# Transformers Provably Learn Chain-of-Thought Reasoning with Length Generalization

**Yu Huang**[*]
Upenn

**Zixin Wen**[*]
CMU

**Aarti Singh**
CMU

**Yuejie Chi**
Yale

**Yuxin Chen**
Upenn

## Abstract

The ability to reason lies at the core of artificial intelligence (AI), and challenging problems usually call for deeper and longer reasoning to tackle. A crucial question about AI reasoning is whether models can extrapolate learned reasoning patterns to solve harder tasks with a longer chain-of-thought (CoT). In this work, we present a theoretical analysis of transformers learning on synthetic state-tracking tasks with gradient descent. Specifically: 1). We prove how the *algebraic structure* of state-tracking problems governs the length generalization of learned reasoning in transformers. In doing so, we formulate the **attention concentration** mechanism, linking the retrieval robustness of the attention layer to the task structure of long-context state tracking problems. 2). Moreover, we prove that a transformer can provably *self-improve* via a **recursive self-training** scheme that progressively extends the range of solvable problem lengths. We show that the model can achieve abilities outside the coverage of the base model in recursive training, different from prior theoretical works on self-improvement. To our knowledge, we provide the first *optimization guarantee* that constant-depth transformers provably learn $NC^1$-complete problems with CoT, significantly going beyond prior art confined in $TC^0$, unless the widely held conjecture $TC^0 \neq NC^1$ fails. Finally, we present a broad set of experiments supporting our theoretical results, confirming the length generalization behaviors and the mechanism of attention concentration.

## 1 Introduction

Reasoning is a central theme of artificial intelligence [1, 2, 3]. Transformer-based [4] large language models (LLMs) achieve state-of-the-art results on complex reasoning tasks via chain-of-thought (CoT) reasoning [5, 6, 7, 8, 9, 10, 11, 12], where the model generates intermediate steps before delivering a final answer. Recent frontier models such as OpenAI-o1 [1] and DeepSeek-R1 [2] typically produce long CoT traces at inference time, often elicited via reinforcement learning and/or supervised fine-tuning (SFT) that distills from longer chains [13, 14]. These advances have enabled improved performance on more challenging problems [15, 16, 17, 18, 19], yet the mechanisms and limitations underlying CoT reasoning remain poorly understood, posing fundamental theoretical challenges.

Theoretical studies on transformers with CoT have recently advanced along two fronts: expressiveness [20, 21, 22, 23, 24] and statistical learnability [25, 26, 27, 28, 29, 30, 31]. On the expressiveness front, seminal work [20, 21, 22] showed that constant-depth transformers without CoT behave as shallow circuits and are restricted to express the circuit complexity class $TC^0$ of constant computation depth, whereas with $O(L)$ CoT steps on inputs of length $L$, they can express log-depth circuits in $NC^1$, which is a problem class conjectured to require inherently serial computation.[1] These

---

[*]The authors contributed equally, and the author order was determined by a coin flip. Given the density of results, we recommend the full version for readability: arXiv:2511.07378

[1] For background on circuit complexity and a detailed review of expressiveness results, see Section 2 and [32, 33].

results reveal that CoT reasoning equips transformers with the expressive power to solve **inherently sequential** problems. In contrast, there remains limited understanding of *how* transformers acquire such reasoning abilities during training. Prior optimization analyses [34, 35, 36, 37] were limited to simple, fully parallelizable tasks in $\mathsf{TC}^0$ that do not require sequential reasoning. This leaves a substantial gap between what transformers can *express* in principle and what they can *learn* through training.

Another key question, motivated by the success of reasoning models, is whether models can extrapolate their reasoning beyond the sequence lengths of the training data: a property known as **length generalization**. In practice, reasoning effectively over long chains of thought relies strongly on the model's *long-context ability* [38, 39, 40, 41], which can be severely affected by *context rot*, a phenomenon in which model performance degrades as the number of tokens in the context increases [42, 43, 44]. Even for synthetic reasoning tasks, empirical evidence on length generalization of transformers is mixed [45, 46, 47, 48, 49, 50, 51], and several prior studies reported limited extrapolation despite strong in-distribution performance. Architectural choices, including positional encoding and attention variants, can influence generalization considerably [52, 53, 54, 55, 56, 57, 58]. On the theoretical front, previous work established existence or statistical guarantees, showing that transformers can, in principle, represent length-generalizing algorithms or achieve favorable sample complexity independent of the length of the CoT [59, 60, 61, 62, 63]. Nonetheless, it remains unclear whether transformers can provably learn to length-generalize reliably when trained with gradient-based optimization algorithms.

In light of the aforementioned gaps in prior literature, the current paper seeks to make progress towards addressing the following two fundamental questions:

> **Research Questions**
>
> *1.* Can transformers, trained via gradient descent (GD), learn CoT reasoning to solve problems requiring *inherently* sequential reasoning beyond $\mathsf{TC}^0$?
>
> *2.* Can the learned reasoning ability *generalize* to problems that require longer CoTs beyond the lengths of training data?

To address these questions in a theoretically tractable manner, we analyze a minimally viable transformer: a one-layer transformer block with softmax attention and a feed-forward network (FFN), trained by GD with *no positional encoding* (NoPE). We study this model on synthetic *state-tracking* tasks, namely LEGO [45], which distill core LLM skills such as entity tracking, game-state updates, and code evaluation [64, 65]. This setup is tractable for analysis, while capturing the mechanisms needed for step-by-step computation via CoT. We analyze the training dynamics with CoT on two LEGO task families with distinct *algebraic* action structures: simply transitive group actions, and symmetry group actions. By tracking attention patterns throughout training, we demonstrate how reasoning capabilities emerge and how length generalization is governed by the structural properties of these tasks. More concretely, our main contributions are summarized as follows.

1. **Provable guarantee of learning CoT with length generalization in state-tracking.** We prove that for the LEGO state-tracking task, a one-layer NoPE transformer trained with GD provably solves constant-length problems using CoT reasoning. Moreover, the learned transformer directly generalizes to problems of substantially longer length, when the group actions in LEGO is *simply transitive*. Conversely, for the canonical action of the symmetry group $S_n$ on $\mathbb{Z}_n$, the learned transformer generalizes only up to constant-factor length. We identify an **attention concentration** mechanism that dictates step-wise retrieval depending on the task structure, leading to distinct length generalization behaviors.

2. **Recursive self-training provably extends the solvable reasoning length.** When length generalization is limited (for example, under symmetry group actions), we introduce a *self-training* curriculum that recursively trains the model on its own CoT traces, motivated by the empirical work [54]. We prove that this scheme can bootstrap the solvable problem length up to maximal allowable length after sufficient rounds of self-training, thus offering a theoretical guarantee of recursive self-improvement.

3. **Constant-depth transformers provably learn to solve problems beyond $\mathsf{TC}^0$ via CoT.** The first two results further establish that the model can learn the solution of state-tracking problems for non-solvable groups, which is $\mathsf{NC}^1$-complete and lies outside $\mathsf{TC}^0$ unless the widely held conjecture $\mathsf{TC}^0 \neq \mathsf{NC}^1$ in circuit complexity theory fails. Therefore, we provide *the first opti-*

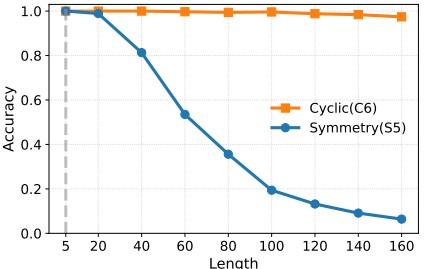
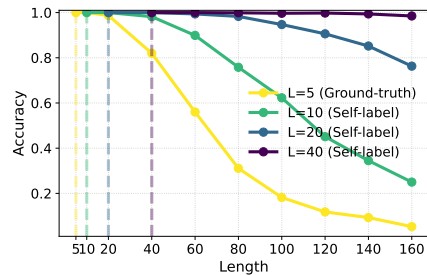

(a) Length generalization results of cyclic ($C_6$) vs. symmetry ($S_5$) tasks.

(b) Self-improvement results on symmetry tasks ($S_5$).

Figure 1: Empirical results of length generalization on LEGO tasks with different group actions. (a). Transformers length-generalize to solve significantly longer CoT tasks for simply transitive (cyclic) group (Theorem 4.1), while generalizing poorly for symmetry group tasks. (b). When direct length generalization falls short for symmetry actions, a recursive self-training scheme that train on the model's own longer CoT traces bootstraps the solvable problem length (Theorem 5.1). The dashed lines indicate the training length.

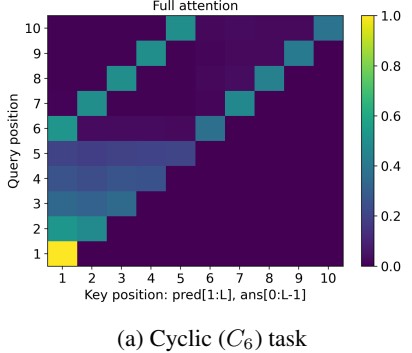
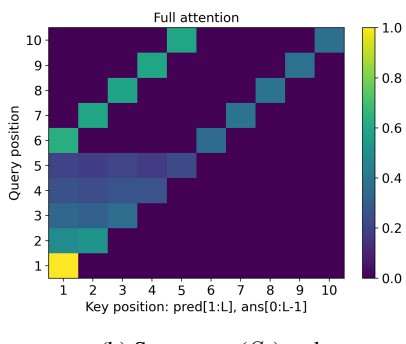

(a) Cyclic ($C_6$) task

(b) Symmetry ($S_5$) task

Figure 2: Attention concentration at convergence for LEGO task with length $L = 5$. The heatmap places the query clause index on the $y$-axis (keys on the $x$-axis). For a task of length $L$, the LEGO sequence prior to the final answer clause has length $2L$; we focus on query positions $L + 1$ to $2L$, corresponding to answer clauses $Z_{\mathsf{ans},0}$ to $Z_{\mathsf{ans},L-1}$. Two diagonal bands in the upper region indicate attention concentrating on the answer clause $Z_{\mathsf{ans},\ell}$ and the predicate clause $Z_{\mathsf{pred},\ell+1}$ when the query is $Z_{\mathsf{ans},\ell}$.

*mization guarantee* showing that a one-layer transformer learns to solve reasoning tasks beyond $\mathsf{TC}^0$ with CoT, matching the expressivity result of [20, 22, 21] for linear-CoT transformers.

4. **Empirical evidence on synthetic LEGO tasks that supports our theory.** We present a wide range of experiments based on our theoretical setup. Our results corroborate our predictions on length generalization for different group actions, showing a clear separation between the two algebraic structures in the theoretical setting. Moreover, we empirically demonstrate how recursive self-training effectively improves reasoning length. The attention concentration mechanism identified in our theory is also supported by our experimental findings.

## 2 Background

Computational complexity [33] has been employed to characterize the power of neural networks [66, 67]. Historically, *circuit complexity* has been used extensively to study the power of neural networks [68, 69, 70, 71, 72, 24, 73], due to the structural resemblance of Boolean circuits and neural networks with threshold gates. In a nutshell, circuit complexity evaluates computation models by the size, depth, and gate types of Boolean circuits that implement them. Below we provide a brief introduction of the basics of circuit complexity.

## 2.1 Circuits and Expressiveness

A Boolean circuit is a finite acyclic network of logic gates that computes a Boolean function on $\{0, 1\}^n \to \{0, 1\}$ for some fixed $n$. The gates with fan-in 0 are the inputs, which are assigned one of the $n$ Boolean variables. A *circuit family* $\{C_n\}_{n \geq 1}$ computes a language by using $C_n$ on length-$n$ inputs. We measure the *size* of a circuit by the number of gates, and *depth* by the length of the longest input-output path, respectively. For instance, "constant depth" means $\mathrm{depth}(C_n) = O(1)$, while "polynomial size" means $\mathrm{size}(C_n) = O(n^c)$ for some integer $c$. Standard circuit complexity classes are defined by restricting circuit depth, size, gate set and fan-in:

- $\mathsf{TC}^0$ consists of *constant-depth*, polynomial-size circuits with unbounded fan-in $\{\mathsf{AND}, \mathsf{OR}, \mathsf{NOT}\}$ gates augmented with threshold (e.g., $\mathsf{MAJORITY}$) gates. The circuit class $\mathsf{TC}^0$ captures exactly the complexity of integer multiplication and division, and sorting [74].
- $\mathsf{NC}^1$ consists of *log-depth*, polynomial-size circuits over $\{\mathsf{AND}, \mathsf{OR}, \mathsf{NOT}\}$ with bounded fan-in. This class captures exactly the complexity of recognizing all regular languages.
- $\mathsf{P/poly}$ is the class of all languages computable by polynomial-size circuit families.

These classes satisfy the following relations:[2]

$$\mathsf{TC}^0 \subseteq \mathsf{NC}^1 \subseteq \mathsf{P/poly}.$$

**Expressiveness gap of transformers via circuit complexity.** One can analyze the expressivity of transformer neural nets through the lens of circuit complexity. Prior work [75, 46, 24, 20, 21] showed that the expressive upper bound of a vanilla constant-depth transformer is limited to $\mathsf{TC}^0$, the class of problems solvable by extremely shallow, highly parallel circuits. In contrast, when equipped with CoT, recent work [20, 22, 21] proved that $O(n)$ intermediate steps enable transformers to simulate $\mathsf{NC}^1$-complete language. Further, [21] showed that 2-layer transformers with polynomially many CoT steps suffice to express arbitrary polynomial-size circuits, i.e., $\mathsf{P/poly}$. Hence, under the widely believed conjecture that $\mathsf{TC}^0 \neq \mathsf{NC}^1$ [32], CoT strictly extends the expressive power of transformers beyond $\mathsf{TC}^0$, allowing them to solve problems that inherently require super-constant (here $O(\log n)$) computation depth. A standard example connecting algebraic structure with circuit complexity is the following *word problem*, define by Dehn [76, 77].

**Definition 2.1** (Word problem for a group $G$). Let $G$ be a group and let $e$ denote its identity. For a word $w = g_1 \cdots g_k \in G^*$ (each $g_i \in G$), the word problem asks to decide whether $g_1 \circ \cdots \circ g_k = e$.

Barrington [78] proved that the above word problem is $\mathsf{NC}^1$-complete when the group is non-solvable.

**Theorem 2.1** (Barrington [78]). *The word problem of every finite non-solvable group is $\mathsf{NC}^1$-complete. The canonical example is $S_n$ for $n \geq 5$, the symmetry group on $n$ elements that encodes the permutations.*

The word problem captures a fundamental reasoning task: state tracking in a finite world [65, 79, 80]. Given an initial state and a sequence of transformations, the goal of the task is to compute the resulting state. State tracking underlies practical LLM abilities such as narrative entity tracking, chess move analysis, and code execution [64, 65], while exhibiting a rich connection to the circuit complexity theory. Therefore, it has become a standard synthetic testbed for probing the reasoning abilities of language models, both theoretically [46, 65, 79] and empirically [45, 46, 80]. Motivated by this connection, we study the state-tracking task with varying algebraic structures.

## 2.2 LEGO for State Tracking

We focus on a specific formulation of the state-tracking problem, LEGO (*Learning Equality and Group Operations*) [45], which was originally proposed as a synthetic task to study the reasoning behavior of transformers empirically. A typical LEGO instance in [45] takes the following form:

```
b = + a,   c = - b,   ...,   t = - s,   a = -1,   ...
```

Here, `a, b, c, ...` are **variables**, each taking a **value** (or state) in $\{-1, +1\}$ in this example. Short expressions such as `b = + a` are **clauses**, where `= +` and `= -` denote **actions**: the action is applied to the *right-hand-side* variable's value to obtain the *left-hand-side* variable's value. For

---

[2] We intentionally omit uniformity conventions here as this section is only meant to give readers a sense of the classes we refer to and their standard relationships.

instance, from `b = + a` and `a = -1`, it follows that `b = -1`. Formally, the LEGO language is defined as follows:

**Definition 2.2** (LEGO language [45]). Let $\mathcal{X}, \mathcal{G}, \mathcal{Y}$ be finite sets of variables, actions, and values, respectively, where each $g \in \mathcal{G}$ is a map $g : \mathcal{Y} \to \mathcal{Y}$. The formal language **LEGO**$(\mathcal{X}, \mathcal{G}, \mathcal{Y})$ has alphabet $\mathcal{X} \cup \mathcal{G} \cup \mathcal{Y} \cup \{=, (,)\}$ and consists of two types of expressions (called **clauses**):

(1) *Predicate clause* $x = g(x')$ specifies an action $g \in \mathcal{G}$ linking variables $x, x' \in \mathcal{X}$.

(2) *Answer clause* $x = y$ assigns a value $y \in \mathcal{Y}$ to a variable $x \in \mathcal{X}$.

A canonical LEGO sentence of length $L$ with answer up to $L'$ concatenates predicate clauses $x_n = g_n(x_{n-1})$ for $n \in [L]$ and answer clauses $x_n = y_n$ for $n \in [L']$ with $L' \leq L$:

$$\underbrace{x_1 = g_1(x_0) \ \ldots\ldots\ x_L = g_L(x_{L-1})}_{\text{predicates}} \underbrace{x_0 = y_0 \ \ldots\ldots\ x_{L'} = y_{L'}}_{\text{answers}}, \tag{1}$$

which describes the chain of transitions:

$$\underbrace{x_0 \xrightarrow{g_1} x_1 \xrightarrow{g_2} x_2 \xrightarrow{g_3} \cdots \xrightarrow{g_{L'}} x_{L'}}_{\text{with answers } y_1, \ldots, y_{L'} \text{ up to } L'} \cdots \xrightarrow{g_L} x_L, \quad \text{starting with } x_0 = y_0.$$

For semantic validity, any sentence containing a path $x_n = g_n(x_{n-1}), \ldots, x_{n-k+1} = g_{n-k+1}(x_{n-k})$ for $k \in [n]$ must satisfy: $y_n = g_n \circ g_{n-1} \circ \cdots \circ g_{n-k+1}(y_{n-k})$.

In the LEGO language, predicate clauses encode transformations, while answer clauses encode observed states. Thus, solving a LEGO sentence is exactly state tracking: compose the listed actions along the path and propagate the observed states to predict the next answer consistent with the composition.

# 3 Problem Formulation

As introduced in Section 2, the current paper employs the LEGO framework to investigate the reasoning capabilities of transformers. To set the stage, this section presents precise mathematical formulations of the problems to be studied in this paper. Before proceeding, we introduce the following notation:

- *Vocabulary.* Define the vocabulary as $\mathcal{V} := \mathcal{X} \cup \mathcal{G} \cup \mathcal{Y} \cup \{\langle \mathsf{blank} \rangle\}$, where the *blank* token $\langle \mathsf{blank} \rangle$ is a null symbol indicating the absence of other tokens.
- *Vocabulary size.* Let $d := |\mathcal{V}|$ denote the (finite) vocabulary size.

To facilitate theoretical analysis, we concentrate on the asymptotic regime where $d \to \infty$. Note that $|\mathcal{X}|, |\mathcal{G}|, |\mathcal{Y}|$ may depend on $d$, and we shall specify any required scaling assumptions as needed.

**Assumption 3.1** (Asymptotic regime). For a language **LEGO**$(\mathcal{X}, \mathcal{G}, \mathcal{Y})$ defined in Definition 2.2, we consider the asymptotic regime where both the vocabulary size $d$ and the number of variables $|\mathcal{X}|$ tend to infinity. Assume $|\mathcal{G}| \leq \log^{C_0} d$ for some constant $C_0 \in [1, 100)$, and hence $\mathcal{Y}$ and $\mathcal{G}$ are much smaller than $\mathcal{X}$ in size.

## 3.1 Data Distribution

We begin by specifying how LEGO clauses are tokenized, followed by a definition of the LEGO distribution.

**Definition 3.1** (LEGO encoding). Each LEGO clause from Definition 2.2 is encoded as a fixed-length, 5-token tuple $Z \in \mathcal{V}^5$. More specifically,

- For each predicate clause $x = g(x')$, set $Z_{\mathsf{pred}} := (x, g, x', \langle \mathsf{blank} \rangle, \langle \mathsf{blank} \rangle) \in \mathcal{V}^5$;
- For each answer clause $x = y$, set $Z_{\mathsf{ans}} := (\langle \mathsf{blank} \rangle, \langle \mathsf{blank} \rangle, \langle \mathsf{blank} \rangle, x, y) \in \mathcal{V}^5$.

With Definition 3.1 in place, we can encode a LEGO sentence (1) to a sequence $Z^{L,L'}$:

$$Z^{L,L'} = (Z_{\mathsf{pred},1}, \ldots, Z_{\mathsf{pred},L}, Z_{\mathsf{ans},0}, \ldots, Z_{\mathsf{ans},L'}), \tag{2}$$

where $Z_{\mathsf{pred},k}$ and $Z_{\mathsf{ans},k}$ represent the $k$-th predicate and answer clauses, respectively. For convenience, we denote $\mathcal{I}^{L,L'} = \{(\mathsf{pred}, \ell)\}_{\ell \in [L]} \cup \{(\mathsf{ans}, \ell)\}_{\ell=0}^{L'}$ as the index set associated with the clauses in $Z^{L,L'}$. If $L = L'$, we write the sequence simply as $Z^L$ and the corresponding index set as $\mathcal{I}^L$.

To feed LEGO tokens into a neural network, we first map each symbol to an integer index (*tokenization*), and then map indices to continuous vectors via a learned table (*embedding*). The following definitions formalize these two steps.

**Definition 3.2** (Tokenization and token embedding). Each $v \in \mathcal{V}$ is assigned a unique index $\tau(v) \in \{1, \ldots, d\}$. Denote by $e_i \in \mathbb{R}^d$ the embedding vector associated with index $i$, and write $e_v \equiv e_{\tau(v)}$ for convenience. The blank token is assigned the zero vector, $e_{\tau(\langle\text{blank}\rangle)} = \mathbf{0}_d \in \mathbb{R}^d$. For technical simplicity, we assume that $\{\, e_v : v \in \mathcal{V} \setminus \{\langle\text{blank}\rangle\} \,\}$ forms an orthonormal set in $\mathbb{R}^d$ (note that this assumption can be relaxed to a well-conditioned embedding matrix without affecting our results).

Equipped with Definition 3.2, we can transform the LEGO sequence encoding into vector embeddings that can be used as inputs to neural networks.

**Definition 3.3** (Embedding of LEGO sentences). Let $d_c := 5d$ be the clause embedding dimension. We define an operator $\mathsf{Embed} : \mathcal{V}^5 \to \mathbb{R}^{d_c}$ that maps a clause to embedding by

$$\mathbf{Z} = \mathsf{Embed}(Z) := (e_{v_1}, e_{v_2}, \ldots, e_{v_5}) \in \mathbb{R}^{d_c}, \quad \text{for clause } Z = (v_1, v_2, \ldots, v_5) \in \mathcal{V}^5.$$

Specifically, a LEGO sentence $Z^{L,L'}$ defined in (2) is embedded as

$$\mathbf{Z}^{L,L'} = (\mathbf{Z}_{\mathsf{pred},1}, \ldots, \mathbf{Z}_{\mathsf{pred},L}, \mathbf{Z}_{\mathsf{ans},0}, \ldots, \mathbf{Z}_{\mathsf{ans},L'}) \in \mathbb{R}^{d_c \times (L+L'+1)}$$

where each column $\mathbf{Z}_{\mathsf{pred},\ell} \in \mathbb{R}^{d_c}$ (resp. $\mathbf{Z}_{\mathsf{ans},\ell}$) is the embedding of clause $Z_{\mathsf{pred},\ell}$ (resp. $Z_{\mathsf{ans},\ell}$). When $L' = L$, we simply write $\mathbf{Z}^L \equiv \mathbf{Z}^{L,L'}$ for simplicity.

Next, we describe the distribution that governs the generation of a LEGO sentence.

**Assumption 3.2** (LEGO distribution $\mathcal{D}^L, \mathcal{D}^{L,L'}$). Consider **LEGO**$(\mathcal{X}, \mathcal{G}, \mathcal{Y})$ as defined in Definition 2.2, and let $L$ denote the sequence length. We assume that the distribution $\mathcal{D}^L$ of length-$L$ LEGO sentences satisfies the following properties.

1. All LEGO sentences $Z^L \sim \mathcal{D}^L$ of the form (1) are encoded by Definition 3.1 into representation (2).

2. The variables $x_0, x_1, \ldots, x_L \in \mathcal{X}$ are sampled uniformly at random from $\mathcal{X}$ without replacement.

3. The first value $y_0 \in \mathcal{Y}$ is chosen uniformly at random from $\mathcal{Y}$.

4. The actions $g_1, g_2, \ldots, g_L \in \mathcal{G}$ are sampled uniformly at random from $\mathcal{G}$ with replacement.

5. The intermediate values $y_1, y_2, \ldots, y_L$ are computed recursively by $y_i = g_i(y_{i-1})$.

For any $L' < L$, we define the truncated distribution $\mathcal{D}^{L,L'}$ of sequences $Z^{L,L'}$ containing all the predicates and the first $L'+1$ many answer clauses, where $Z^{L,L'}$ is obtained by first sampling $Z^L \sim \mathcal{D}^L$ and then removing the answer clauses $Z_{\mathsf{ans},\ell}$ for all $\ell > L'$.

As can be easily seen, sequences sampled from $\mathcal{D}^L$ or $\mathcal{D}^{L,L'}$ correspond to valid LEGO sentences as defined in Definition 2.2. With a slight abuse of notation, we write $\mathbf{Z}^{L,L'} \sim \mathcal{D}^{L,L'}$ to indicate that $\mathbf{Z}^{L,L'}$ is the embedding of a sentence $Z^{L,L'}$ drawn from $\mathcal{D}^{L,L'}$.

## 3.2 Transformer Architecture

In this subsection, we introduce the transformer architecture investigated in this paper. Towards this end, we first introduce a smoothed activation function that will be used in our network.

**Definition 3.4** (Smooth ReLU). Define a continuously differentiable variant of the ReLU activation function [81, 82] as follows:

$$\mathbf{sReLU}(x) := \begin{cases} \dfrac{\varrho}{q}, & x \leq -\varrho, \\ \dfrac{x^q}{\varrho^{q-1}q}, & x \in (-\varrho, \varrho], \\ x - \varrho\left(1 - \dfrac{1}{q}\right), & x > \varrho, \end{cases}$$

where $q$ and $\varrho$ are some design parameters. Here and throughout, we choose $q = O(1)$ to be a large, even integer, and take $\varrho = \Theta(1/\mathrm{polylog}(d))$.

**Transformer architecture.** In this work, we focus on an autoregressive transformer [4] whose block consists of a softmax attention layer followed by a position-wise feed-forward network, as described below.

- *Attention layer.* Given LEGO sentence embeddings $\mathbf{Z}^{L,L'}$ and indices $\boldsymbol{j}, \boldsymbol{k} \in \mathcal{I}^{L,L'}$, the attention from clause $\mathbf{Z}_{\boldsymbol{j}}$ to clause $\mathbf{Z}_{\boldsymbol{k}}$ is defined through the softmax operator as

$$\mathbf{Attn}_{\boldsymbol{j} \to \boldsymbol{k}}(\mathbf{Q}, \mathbf{Z}^{L,L'}) := \frac{\exp(\mathbf{Z}_{\boldsymbol{j}}^{\top} \mathbf{Q} \, \mathbf{Z}_{\boldsymbol{k}})}{\sum_{\boldsymbol{r} \in \mathcal{I}^{L,L'}} \exp(\mathbf{Z}_{\boldsymbol{j}}^{\top} \mathbf{Q} \, \mathbf{Z}_{\boldsymbol{r}})}.$$

  Since the model is autoregressive, a standard causal mask is applied to ensure that the latest (answer) token attends only to *preceding* tokens (including itself). The attention output is

$$\text{Attention}(\mathbf{Q}, \mathbf{Z}^{L,L'}) := \sum_{\boldsymbol{k} \in \mathcal{I}^{L,L'}} \mathbf{Attn}_{\text{ans}, L' \to \boldsymbol{k}}(\mathbf{Q}, \mathbf{Z}^{L,L'}) \cdot \mathbf{Z}_{\boldsymbol{k}}.$$

  Note that in the standard formulation of transformers, the score takes the form $\mathbf{Z}_{\boldsymbol{j}}^{\top} \mathbf{W}^{Q\top} \mathbf{W}^{K} \mathbf{Z}_{\boldsymbol{k}}$ instead for a "query" parameter matrix $\mathbf{W}^{Q}$ and a "key" parameter matrix $\mathbf{W}^{K}$. Here, we fold $\mathbf{W}^{Q\top} \mathbf{W}^{K}$ into a single parameter matrix $\mathbf{Q}$, resulting in a score $\mathbf{Z}_{\boldsymbol{j}}^{\top} \mathbf{Q} \mathbf{Z}_{\boldsymbol{k}}$, which can be viewed as an equivalent reparameterization that simplifies analysis without changing expressivity [83, 84, 85].

- *Feed-forward network (FFN).* The FFN with parameter $\mathbf{W} \in \mathbb{R}^{5 \times d \times m \times d_c}$ is defined by

$$\text{FFN}_{i,j}(\mathbf{W}, \mathbf{X}) := \sum_{r \in [m]} \mathbf{sReLU}\Big(\langle \mathbf{W}_{i,j,r}, \mathbf{X} \rangle + b_{i,j,r}\Big), \quad \forall\, i \in [5],\, j \in [d],$$

  where $\mathbf{X}$ denotes the input, $\mathbf{W}_{i,j,r} \in \mathbb{R}^{d_c}$ are neuron weights, $m$ indicates the number of neurons, and $b_{i,j,r} \in \mathbb{R}$ is some *fixed* bias.

With the above layers defined, we are ready to introduce the transformer model used in this work, which is summarized as follows.

**Definition 3.5** (Transformer language model). Assume that our learner neural network $F$ is a one-layer decoder transformer block composed of an attention layer with no positional encoding (NoPE) as well as an FFN layer: $F = \text{FFN} \circ \text{Attention}$, with parameters $\mathbf{W} \in \mathbb{R}^{5 \times d \times m \times d_c}$ and $\mathbf{Q} \in \mathbb{R}^{d_c \times d_c}$. Formally, for the $i$-th token position and the $j$-th vocabulary index, we have

$$\Big[F_i\big(\mathbf{Z}^{L,L'}\big)\Big]_j := \text{FFN}_{i,j}\Big(\mathbf{W}, \text{Attention}\big(\mathbf{Q}, \mathbf{Z}^{L,L'}\big)\Big) \in \mathbb{R}, \quad \forall\, i \in [5], j \in [d]. \tag{3}$$

We interpret $F(\mathbf{Z}^{\ell})$ as five logit vectors $\{F_i(\mathbf{Z}^{\ell})\}_{i=1}^{5} \subset \mathbb{R}^d$, each parameterizing the distribution of the $i$-th token of the next clause. Recall that $\mathcal{V}$ is the vocabulary with $|\mathcal{V}| = d$ and $\tau : \mathcal{V} \to [d]$ denotes the index map. Given an encoded LEGO sequence $Z^{\ell} = (Z_1, \ldots, Z_{\ell})$ with embedding $\mathbf{Z}^{\ell}$, the model's predictive distribution for the $i$-th token of the $(\ell + 1)$-th clause is the following softmax:

$$p_{F_i}\big(Z_{\ell+1,i} = v \mid Z_1, \ldots, Z_{\ell}\big) = \frac{e^{[F_i(\mathbf{Z}^{\ell})]_{\tau(v)}}}{\sum_{j \in [d]} e^{[F_i(\mathbf{Z}^{\ell})]_j}}, \quad \forall\, v \in \mathcal{V}. \tag{4}$$

Now we can sample the next clause $Z_{\ell+1}$ by sampling from the product distribution

$$Z_{\ell+1} = (Z_{\ell+1,1}, \ldots, Z_{\ell+1,5}) \sim \bigotimes_{i=1}^{5} p_{F_i} =: p_F,$$

in an autoregressive manner.

### 3.3 LEGO Task via CoT Reasoning and Training Objective

With the LEGO distribution and the transformer model in place, we now formalize the task within the LEGO framework. We view solving a length-$L$ LEGO problem as generating a sequence of CoT steps, where each step $\ell$ produces an intermediate result used to predict $x_\ell = y_\ell$, ultimately leading to the final solution $x_L = y_L$. To be precise, we define the following reasoning task.

**Definition 3.6** (Reasoning tasks $\mathcal{T}^L$). We define a family $\{\mathcal{T}^L\}_{L \in \mathbb{N}^+}$ that captures the ability to solve sequential reasoning problems. For each $L \geq 1$, task $\mathcal{T}^L$ measures the model's *accuracy along*

*the chain* from step 1 to step $L$:

$$\text{Acc}_L(F) = \frac{1}{L} \sum_{0 \leq L' < L} \mathbb{E}_{Z^L \sim \mathcal{D}^L} \left[ \mathbb{E}_{\widehat{Z}_{\text{ans}, L'+1} \sim p_F(\cdot | Z^{L,L'})} \left[ \mathbb{1}\{\widehat{Z}_{\text{ans}, L'+1} = Z_{\text{ans}, L'+1}\} \right] \right], \quad (5)$$

where $p_F$ is induced by the model $F$ from (4).

Clearly, $\text{Acc}_L(F) \in [0, 1]$. We say that task $\mathcal{T}^L$ is solved if $\text{Acc}_L(F) \approx 1$. As $L$ grows, $\{\mathcal{T}^L\}_{L \in \mathbb{N}^+}$ poses increasingly difficult state tracking challenges. At step $L'$, the model conditions on the partial transcript $Z^{L,L'}$ and predicts the next answer $Z_{\text{ans}, L'+1}$. To enforce step-by-step generation of the CoT trace, we define the following training objective.

**Definition 3.7** (Next clause loss). The training objective for $\mathcal{T}^L$ with $L \geq 1$ is the *next clause* loss

$$\text{Loss}^L(F) := \sum_{1 \leq L' \leq L} \text{Loss}^{L,L'}(F), \quad (6a)$$

$$\text{where} \quad \text{Loss}^{L,L'}(F) := \mathbb{E}_{Z^{L,L'} \sim \mathcal{D}^{L,L'}} \left[ -\log p_F(Z_{\text{ans}, L'} \mid Z^{L,L'-1}) \right]. \quad (6b)$$

Here, $p_F$ is induced by the model $F$ from (4). We also define the per-token loss

$$\text{Loss}_i^{L,L'}(F) := \mathbb{E}_{Z^{L,L'} \sim \mathcal{D}^{L,L'}} \left[ -\log p_{F_i}(Z_{\text{ans}, L', i} \mid Z^{L,L'-1}) \right].$$

This is a teacher forcing style CoT objective: the model is given the ground truth answers for all previous steps and is trained to match the next step's answer [86, 34].

Further, we adopt the following initialization for training.

**Assumption 3.3** (Initialization). Let $F$ be the transformer network in Definition 3.5 with parameters $\mathbf{W}$ and $\mathbf{Q}$. The attention parameter is initialized to zero: $\mathbf{Q}^{(0)} = \mathbf{0}_{d_c \times d_c}$. The FFN weights are initialized randomly and independently as $\mathbf{W}_{i,j,r}^{(0)} \sim \mathcal{N}(\mathbf{0}, \sigma_0^2 \mathbf{I}_d)$ with $\sigma_0 = d^{-1/2}$. The biases are not trained and instead fixed at $b_{i,j,r} = \sigma_0 \log d$ for all $i, j, r$, chosen to keep most sReLU units active at initialization. All random draws are independent across indices.

# 4 Learning CoT on Simply Transitive Actions

In this section, we present our first main results for the case where the group action $\mathcal{G}$ on $\mathcal{Y}$ is simply transitive, which is isomorphic to the action of the cyclic group $C_n$ on $\mathbb{Z}_n$.

**Assumption 4.1** (Simply transitive group action). Let $\mathcal{Y} = \{0, 1, \ldots, n_y - 1\}$ with $n_y \in [\Omega(\log \log d), \log d]$. Assume the group $\mathcal{G}$ acts simply transitively on $\mathcal{Y}$: the action is transitive and for any $y_1, y_2 \in \mathcal{Y}$ there exists a unique $g \in \mathcal{G}$ such that $g \cdot y_1 = y_2$. Equivalently, for any fixed $y \in \mathcal{Y}$ the map $g \mapsto g \cdot y$ is a bijection from $\mathcal{G}$ to $\mathcal{Y}$, so $|\mathcal{G}| = |\mathcal{Y}| = n_y$.

Our first main result demonstrates that, for such a simple task the model obtained via Algorithm 1 for short-chain tasks $\mathcal{T}^1$ and $\mathcal{T}^2$, successfully generalizes to significantly longer tasks.

**Theorem 4.1.** *Under Assumptions 3.1 to 3.3, 4.1, C.1 and C.2, for some constant $0 < c^* < 1$, $n_y < m \ll \log^2 d$, the transformer model $F^{(T_1+T_2)}$ obtained by Algorithm 1 with learning rate $\eta = \frac{1}{\text{poly}(d)}$, and stage 1 and 2 iteration $T_1 = \widetilde{O}\left(\frac{d}{\eta(\sigma_0)^{q-2}}\right)$, $T_2 = \widetilde{O}\left(\frac{\text{poly}(d)}{\eta \sigma_0}\right)$ satisfies*

1. **Direct short-to-long length generalization:**

$$\text{Acc}_L\left(F^{(T_1+T_2)}\right) \geq 1 - \frac{1}{\text{poly}(d)}, \text{ for every } L \leq O(d^{c^*}), \quad (7)$$

*i.e., $F^{(T_1+T_2)}$, which is trained for task $\mathcal{T}^1$ and $\mathcal{T}^2$, generalizes to solve the tasks $\mathcal{T}^\ell, \ell \leq L$.*

2. **Attention concentration:**[3] *given $Z^{2,\ell}$ with $\ell \in \{0, 1\}$, we have*

$$\mathbf{Attn}_{\text{ans}, \ell \to \text{pred}, \ell+1}^{(T_1+T_2)} + \mathbf{Attn}_{\text{ans}, \ell \to \text{ans}, \ell}^{(T_1+T_2)} \geq 1 - O\left(\frac{1}{d^{c^*}}\right). \quad (8)$$

---

[3] For readability, we abbreviate $\mathbf{Attn}_{\mathbf{k} \to \mathbf{k}'}(\mathbf{Q}^{(t)}, \mathbf{Z})$ by $\mathbf{Attn}_{\mathbf{k} \to \mathbf{k}'}^{(t)}$, omitting explicit dependence when immaterial.

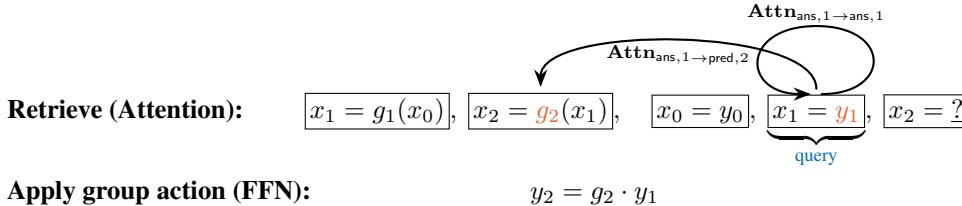

**Retrieve (Attention):** $\boxed{x_1 = g_1(x_0)}, \boxed{x_2 = \textcolor{orange}{g_2}(x_1)}, \quad \boxed{x_0 = y_0}, \boxed{x_1 = \textcolor{orange}{y_1}}, \boxed{x_2 = \underset{-}{?}}$

**Apply group action (FFN):** $\qquad\qquad y_2 = g_2 \cdot y_1$

Figure 3: Illustration of how the model solves the LEGO task: given $Z^{2,1}$, the goal is to predict $y_2$.

**Mechanism of CoT for state-tracking.** Given the current sequence $Z^{L,\ell}$ with intermediate steps up $\ell$, predicting the next state $y_{\ell+1} = g_{\ell+1}(y_\ell)$ requires two steps:

(i) **Retrieve** the correct action $g_{\ell+1}$ from the context clause $Z_{\mathsf{pred},\ell+1}$ and the current state $y_\ell$ from the answer clause $Z_{\mathsf{ans},\ell}$;

(ii) **Apply the group action**, that is, compute the next state $y_{\ell+1} = g_{\ell+1}(y_\ell)$.

It is well established that attention can implement content-based retrieval [87], and that FFN can represent the group operation [88] (see Figure 3 for an example). Algorithm 1 decouples learning the attention (retrieval) and FFN (action) components to simplify the analysis. For task $\mathcal{T}^1$, the transcript $Z^{1,0}$ contains only the two relevant clauses, $Z_{\mathsf{pred},1}$ and $Z_{\mathsf{ans},0}$, without useless contents. Fixed uniform attention (**Q** initialized to be zero in Assumption 3.3) therefore suffices to expose both clauses to the FFN, and we optimize the FFN to learn the group operation. The high accuracy for $\mathcal{T}^1$ in Theorem 4.1 indicates that the FFN has indeed *learned* to apply the operation correctly. For task $\mathcal{T}^2$, with the FFN already trained, the attention layer only needs to learn to route the correct context to the FFN input. The attention concentration result in (8) confirms that the learned routing pattern is correct. Figures 2 and 4 also demonstrate the attention concentration patterns empirically.

**How does attention concentration induce strong length generalization?** As we increase the chain length in $\mathcal{T}^\ell$ for $\ell > 2$, the FFN layer remains largely insensitive to input length since the learned group action is location-invariant. By contrast, the attention layer is affected: more *irrelevant* clauses appear, so retrieval must scan over longer contexts, which risks diluting attention on the relevant clause. Theorem 4.1 guarantees that training on short chains already yields attention concentration with error $O(d^{-c^*})$. This "purity" allows the model to tolerate dilution and maintain high attention on the relevant clauses for chain lengths up to $O(d^{c^*})$. Technically, this concentration arises because the query matrix **Q** learns to locate the same variable $x_\ell$ that appears simultaneously: the third token of the context clause $Z_{\mathsf{pred},\ell+1}$ and the fourth token of the answer clause $Z_{\mathsf{ans},\ell}$. This co-occurrence furnishes a strong, consistent signal that enables robust retrieval across longer chains, which will be elaborated in the proof overview in Appendix C.2.

## 5 Learning CoT on Symmetry Group Actions on $\mathbb{Z}_n$

We now turn to the case where the action group $\mathcal{G}$ is isomorphic to the symmetry group, under Assumption 5.1. In this case, the problem is $\mathsf{NC}^1$-complete for $n_y \geq 5$.

**Assumption 5.1** (Symmetry group actions). Let $\mathcal{Y} = \{0, 1, \dots, n_y - 1\}$. We set $\mathcal{G} = \mathbf{Sym}(\mathcal{Y})$, the symmetry group of all permutations of $\mathcal{Y}$, so that $|\mathcal{G}| = n_y!$. We let $\mathcal{G}$ act on $\mathcal{Y}$ in the natural way and write $g \cdot y$ (or $g(y)$) for the image of $y \in \mathcal{Y}$ under $g \in \mathcal{G}$. We assume $n_y = \Theta\left(\frac{\log \log d}{\log \log \log d}\right)$, and hence $|\mathcal{G}| = n_y! = \mathrm{polylog}d$.

**Challenges of learning CoT for the symmetry task.** When the model tries to retrieve the group element $g_{\ell+1}$ from the correct context clause $Z_{\mathsf{pred},\ell}$, there can be other context clauses whose group elements also send $y_\ell$ to $y_{\ell+1}$; we call these *distractor* clauses. In the symmetry case on $\mathcal{Y}$, each pair $(i, j)$ admits $(n_y - 1)!$ elements mapping $i$ to $j$, so the fraction of distractors is substantial. By contrast, in the simply transitive setting each pair has a unique element, so distractors are unlikely and can be ignored. Attending to distractors still produces the correct next answer, so training may converge with **insufficient attention concentration**. This weaker concentration makes the attention layer less robust to dilution in longer contexts. Hence, for this harder setting, directly proving $d^{\Omega(1)}$ length CoT generalization from constant-length training is difficult.

**Self-improvement for reasoning length extension.** Recent empirical studies [89, 90, 54] show that *length generalization* can be bootstrapped via model *self-improvement*: models training on their own

output can bootstrap their capability to solve longer problems. In particular, the work [54] motivates a **recursive self-training** scheme for the symmetry task. To perform recursive self-training, we adopt the greedy language model as data annotator: the greedy language model $\widehat{p}_F$ induced by the network $F$ is defined by

$$\widehat{p}_F(Z_{\mathsf{ans},L'+1}|Z^{L,L'}) = \begin{cases} 1, & \text{if } Z_{\mathsf{ans},L'+1} = \mathrm{argmax}_Z\, p_F(Z|Z^{L,L'}), \\ 0, & \text{otherwise.} \end{cases} \tag{9}$$

Now we can define the self-annotated LEGO data distribution:

**Definition 5.1** (Bootstrapped LEGO distribution). We define $\mathcal{D}_F^{L,L'}$ as the LEGO distribution in Assumption 3.2 except that the answers $Z_{\mathsf{ans},\ell}, 1 \leq \ell \leq L'$ is given recursively by sampling the prediction $Z_{\mathsf{ans},\ell} \sim \widehat{p}_F(\cdot|Z^{L,\ell-1}), 1 \leq \ell \leq L'$ from the greedy language model $\widehat{p}_F$.

**Definition 5.2** (Self-training loss). Given a (fixed) model $\widetilde{F}$ and length $L$, The self-training next-clause-prediction loss is defined by replacing $\mathcal{D}^{L,L'}$ with $\mathcal{D}_{\widetilde{F}}^{L,L'}$ (Definition 5.1) in (6):

$$\mathsf{Loss}_{\widetilde{F}}^{L,L'}(F) \triangleq \mathbb{E}_{Z^{L,L'}\sim\mathcal{D}_{\widetilde{F}}^{L,L'}} \left[ -\log p_F(Z_{\mathsf{ans},L',i} \mid Z^{L,L'-1}) \right], \tag{10a}$$

$$\mathsf{Loss}_{\widetilde{F},i}^{L,L'} \triangleq \mathbb{E}_{Z^{L,L'}\sim\mathcal{D}_{\widetilde{F}}^{L,L'}} [-\log p_{F_i}(Z_{\mathsf{ans},L',i} \mid Z^{L,L'-1})] \quad \text{for } i \in [5]. \tag{10b}$$

We now present our main results, establishing that a recursive self-training scheme can provably bootstrap the reasoning length for the symmetry LEGO task.

**Theorem 5.1.** *Assume the distribution $\mathcal{D}^L$ induced from **LEGO**$(\mathcal{X}, \mathcal{G}, \mathcal{Y})$ satisfies Assumption 3.1, 3.2 and 5.1, and assume the transformer network satisfies Assumption 3.3, C.1, C.2, and $n_y < m \ll \log^2 d$. Then for any $1 \leq k < \log_2 |\mathcal{X}|$, the transformer $F^{(T_k)}$ trained via Algorithm 2 up to length $L_k = 2^k$ and $T_k = O(\frac{\mathsf{poly}(d)}{\eta})$ satisfies:*

1. **Constant-factor length generalization:** $F^{(T_k)}$ *is able to solve* $\mathcal{T}^{L_{k+1}}$ *with* $L_{k+1} = 2^{k+1}$

$$\mathrm{Acc}_{L_{k+1}}\left(F^{(T_k)}\right) = 1 - \frac{1}{\mathsf{poly}(d)}. \tag{11}$$

2. **Attention concentration:** *given* $Z^{L_k,\ell}$ *with* $\ell \in \{0, \ldots, L_k - 1\}$, *we have*

$$\mathbf{Attn}_{\mathsf{ans},\ell\to\mathsf{pred},\ell+1}^{(T_k)} + \mathbf{Attn}_{\mathsf{ans},\ell\to\mathsf{ans},\ell}^{(T_k)} \geq 1 - \widetilde{c}, \tag{12}$$

   *where $\widetilde{c}$ is some sufficiently small constant (smaller than 0.01).*

At convergence for the current task $\mathcal{T}^{L_k}$ (at time $T_k$, the loss has fallen below $1/d^{\mathsf{E}_1}$ in Algorithm 2), (12) confirms that attention concentration is still insufficient. Nevertheless, Theorem 5.1 shows that while this level of concentration cannot withstand the dilution from much longer contexts, it is sufficient for doubling the length. Consequently, a model trained progressively on $\mathcal{T}^L$ for $L = 1, 2, \ldots, 2^k$ generalizes to the more challenging task of length $2^{k+1}$, yielding the following corollary.

**Corollary 5.1** (Self-improvement for $|\mathcal{X}|$-length reasoning). *Under the same assumptions as Theorem 5.1, letting $K = \Theta(\log d)$, for any length $L \leq |\mathcal{X}|$, the model $F^{(T_K)}$ trained via Algorithm 2 achieves*

$$\mathrm{Acc}_L\left(F^{(T_K)}\right) \geq 1 - \frac{1}{\mathsf{poly}(d)}.$$

**Significance of the result.** Note that $\{x_\ell\}_{\ell=0}^L$ is sampled from $\mathcal{X}$ *without replacement* (Assumption 3.2), the longest feasible chain scales with the variable size: $L+1 \leq |\mathcal{X}| = \Theta(d)$. Thus our guarantee attains the best possible length in this setting. Corollary 5.1 also demonstrates that the transformer can be trained to solve a task beyond $\mathsf{TC}^0$ with linear-step CoT, matching the expressivity result of [21].[4] While prior empirical work reports self-improvement in practice, theoretical guarantees, especially for transformers and length generalization, have been scarce [91, 92, 93]. Theorem 5.1 provides, to our knowledge, the first rigorous evidence that transformers can *bootstrap* their reasoning via self-training without additional supervision.

---

[4]    There is a caveat that we do not study when the embedding dimension is logarithmic in the problem length, although it is possible if we allow over-complete basis for the tokens in $\mathcal{X}$ only.

## Acknowledgments

The work of Z. Wen is supported in part by NSF DMS-2134080 and DMS-2134133. Y. Chen is supported in part by the Alfred P. Sloan Research Fellowship, the NSF grants IIS-2218713, IIS-2218773, and CIF-2221009, the ONR grants N00014-22-1-2354 and N00014-25-1-2344, the Wharton AI & Analytics Initiative's AI Research Fund, and the Amazon Research Award. Z. Wen thanks Yuanzhi Li for initial contributions and helpful discussions on this project.

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

# Appendix: Experiments and Complete Proofs

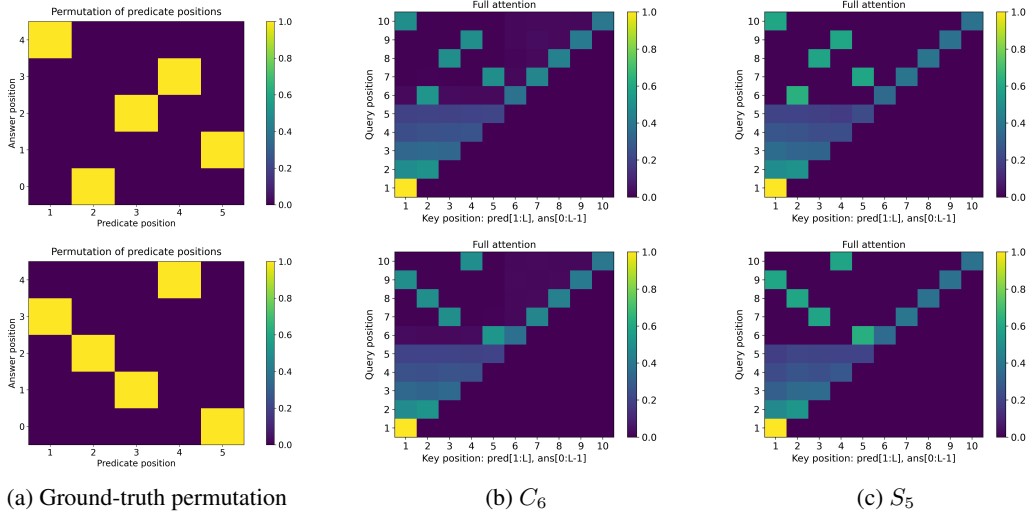

(a) Ground-truth permutation          (b) $C_6$          (c) $S_5$

Figure 4: Attention patterns of the same trained model as Figure 2, evaluated with randomly permuted predicate-clause positions. Column (a) gives the ground-truth permutation. Column (b) and Column (c) show the attention heatmaps for the simply transitive and symmetry tasks, respectively.

# A  Experiments

In this section, we conduct synthetic experiments to verify our theoretical claim.

**General Setup.** We adopt an experimental setting closely aligned with our theoretical setup. Data strictly follow the LEGO distribution in the five-token-per-clause format defined in Definition 3.1 and Assumption 3.2. The simply transitive task uses the cyclic group of order 6 (denoted $C_6$), and the symmetry task uses the symmetry group on five elements (denoted $S_5$). The network is a one-layer, decoder-only transformer with two attention heads and a FFN block. Training optimizes the next-clause loss in (6a) using Adam [94] with a learning rate of 1e-4. We train for 300 epochs to ensure the training loss approaches zero and the model converges. For evaluation, to measure the transformer's ability to solve CoT reasoning tasks, we eschew the teacher-forced accuracy in (5), which conditions on the ground-truth answer prefix, and instead evaluate final-answer accuracy after autoregressively generating all intermediate answer steps without teacher forcing. This better mimics CoT reasoning and is more challenging because errors can accumulate during self-rollout. For computational efficiency, we report the probability that the model predicts the value of all answer steps correctly, following [21].

**Length generalization for different group actions.** Figure 1a shows that when trained at the short length $L = 5$, the model exhibits strong length generalization, achieving near-perfect accuracy at much longer lengths, for example, up to $L = 160$ on the simply transitive task, which corroborates Theorem 4.1. By contrast, on the harder symmetry task the generalization is weaker, yielding only constant-factor extensions as expected.

**Self-training improves reasoning length.** To extend the solvable length on the harder symmetry task, we adopt a recursive self-training curriculum inspired by [54]. We first train at $L = 5$ with ground-truth answer supervision until convergence. At the next stage, we double the length to $L = 10$, use the $L=5$ model to greedily generate answer traces (self-labels) for the $L=10$ data, and retrain on these pseudo-labels. We repeat this doubling process for three stages ($L = 5 \rightarrow 10 \rightarrow 20 \rightarrow 40$), so that the final model is trained on self-labeled data at $L = 40$. Figure 1b reports the length-generalization curve for each stage, with the dashed line (matching the curve's color) marking that stage's training length. Across stages, we observe at least constant-factor generalization beyond the training length; after multiple rounds of self-training, the model achieves nearly perfect accuracy at lengths far exceeding the initial $L = 5$, ultimately matching the simply transitive task's performance up to $L = 160$. These results validate Theorem 5.1 and demonstrate the effectiveness of recursive self-training for extending reasoning length.

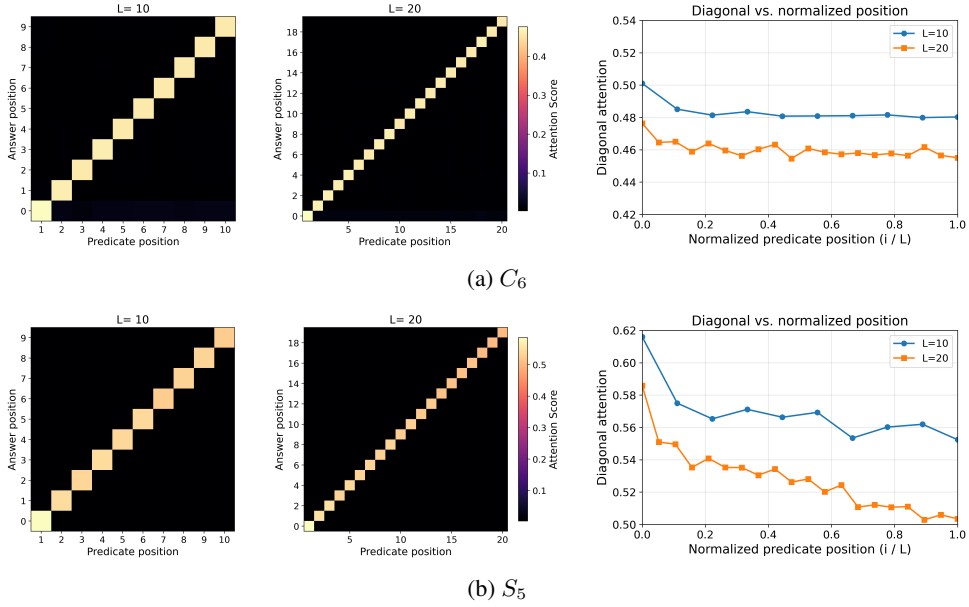

(a) $C_6$

(b) $S_5$

Figure 5: Predicate-clause attention for models trained with LEGO task of length $L = 5$ and tested with $L = 10, 20$. We extract the heatmap of the attention from answer queries to predicate clauses (the upper-left part of the full attention heatmap as in Figure 2), and line-plot the diagonal attention against the normalized predicate position. At longer contexts, the symmetry group task displays a visible drop in attention to the target predicate $Z_{\mathsf{pred}, \ell+1}$ at later predicate locations (larger $\ell$; note the absolute query index $L+\ell$ also increases). By contrast, the cyclic group task maintains more consistent attention across lengths and positions.

**Attention concentration.** We visualize attention heatmaps at convergence for models trained at length $L = 5$ on both simply transitive and symmetry group tasks. Each attention matrix is averaged across the two heads and over 100 independently sampled LEGO sequences. Note that for a task of length $L$, the LEGO sequence prior to the final answer clause has length $2L$. We focus on the query positions $L+1, \dots, 2L$, which correspond to the answer clauses $Z_{\mathsf{ans},0}, \dots, Z_{\mathsf{ans},L-1}$ for predicting all outputs $y_1, \dots, y_L$.

- In Figure 2, the two diagonal structures in the upper region indicate attention concentrating on the answer clause $Z_{\mathsf{ans},\ell}$ and on the predicate clause $Z_{\mathsf{pred},\ell+1}$ when the query is $Z_{\mathsf{ans},\ell}$, validating the attention concentration principle highlighted by Theorem 4.1 and Theorem 5.1.

- To ablate positional bias, we test on the trained model from Figure 2 and only change the input format by randomly permuting the locations of predicate clauses. Results in Figure 4 show that retrieval is keyed to the shared variable rather than absolute position.

- Figures 5 and 6 probe out-of-length (OOL) behavior through attention patterns at test lengths $L \in \{10, 20\}$. As the line plot indicates, the attention scores for both the target predicate and the answer clauses decrease as the task length increases from 10 to 20. This pattern matches the attention dilution predicted by our theory of length generalization. In Figure 5, for the symmetry group task, attention to the correct predicate $Z_{\mathsf{pred},\ell+1}$ decays with position, becoming weaker at later predicates (larger $\ell$, hence a larger absolute index $L+\ell$). By contrast, the cyclic group task shows no comparable decay, with attention remaining fairly uniform across lengths and positions. In Figure 6, the pattern reverses: attention over answer clauses is more stable for the symmetry group, whereas the cyclic case is more sensitive to length and location. We conjecture that length generalization in the LEGO task hinges on a robust predicate–attention pattern. Accordingly, the observed length-dependent decay of attention on predicate clauses in the symmetry task indicates a non-robust learned attention mechanism, which may underlie its poorer length generalization.

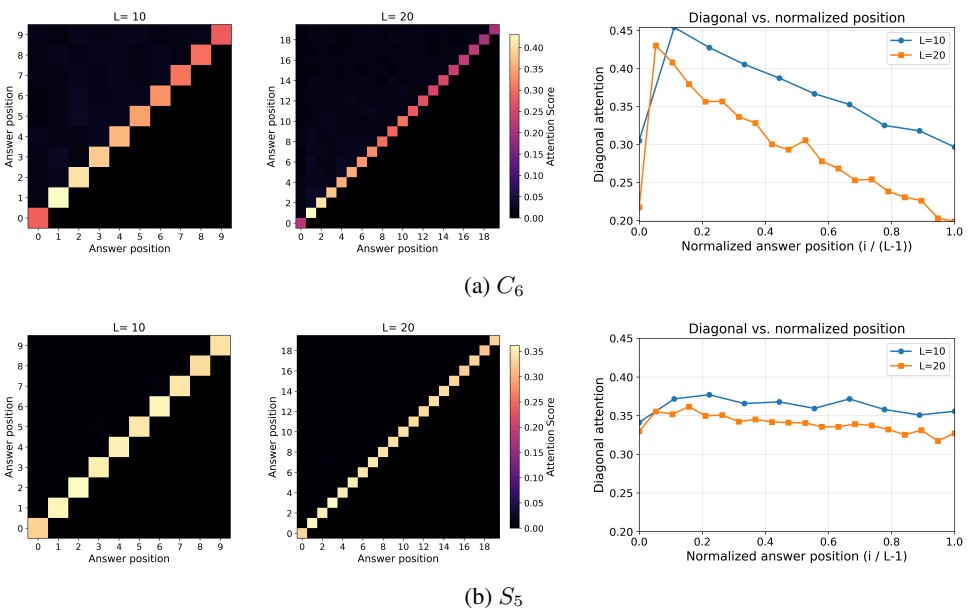

(a) $C_6$

(b) $S_5$

Figure 6: Answer-clause attention for models trained with LEGO task of length $L = 5$ and tested with $L = 10, 20$. We extract the heatmap of the attention from answer queries to answer clauses (the upper-right part of the full attention heatmap, as in Figure 2), and line-plot the diagonal attention against the normalized answer position. In contrast to the predicate attention in Figure 5, at longer contexts, the symmetry group task preserves a stable attention for the answer clause, while the answer attention pattern of the cyclic group task is sensitive to the lengths and positions. Nevertheless, the sharp decay of attention scores for the cyclic group task did not result in a visible performance drop for length generalization, as seen in Figure 1.

## B    Discussions

**NoPE, recency bias, and length generalization.**  Empirically, standard positional embeddings often hinder length extrapolation, while NoPE has been advantageous considering its length-generalization performance [52, 95]. Prior works [34, 35, 37] typically adopt fixed positional encodings, which tie the learned computation to the training horizon. Intuitively, positional embeddings inject location-specific biases that favor local neighborhoods, making longer inputs harder. In our setting, length generalization is possible because the model retrieves relevant information from long, unordered contexts by *content* (variables) rather than by position, as discussed in Appendix C.2 and verified empirically in Figure 4. This provides concrete architectural guidance for practice and identifies positional embeddings as a plausible cause of observed failures to generalize to unseen lengths.

**Local aggregation of tokens is beneficial.**  While NoPE is appealing for length generalization, it lacks position-aware horizontal mixing within a layer, which often hurts empirical performance. In practice, many strong models adopt RoPE (Rotary Positional Encoding) [96, 97, 98], and recent seminal work [95] introduces the *Cannon* layer, a short convolutional window that adds local residual links across neighboring tokens, to inject locality. Integrating Cannon layers substantially boosts the reasoning depth of NoPE transformers in synthetic experiments. Our clause-based data can be interpreted as mimicking the *output* of a Cannon layer: locality is built directly into the data structure, enabling one-layer transformers to learn induction heads, which otherwise require at least a two-layer transformer to express.

**Context rot.**  Context rot is a phenomenon that the performance of LLMs drops as their input context grows even when the task is unchanged, which has been widely noted in practice [43]. A prevailing empirical view is that longer contexts introduce many *distractors* and other irrelevant tokens, forcing the relevant signal to compete with them for attention and thereby weakening retrieval. Our attention concentration perspective characterizes this competition at a fine-grained level: we show that as irrelevant clauses accumulate, attention mass is diluted away from the correct clause, reducing performance at extended reasoning lengths. This offers a simple theoretical perspective on the origin of context rot and why long-context training hasn't been as effective in eliminating

---

**Algorithm 1:** Curriculum training for simply transitive actions

---

**Input:** Model $F^{(0)}$ with parameters $(\mathbf{W}^{(0)}, \mathbf{Q}^{(0)})$; Learning rate $\eta$; Stage snapshots $T_1, T_2$.

**Stage 1:** Learning one-step reasoning ($\mathcal{T}^1$) ;

    **for** $t = 1$ **to** $T_1$ **do**                       `// Update the FFN parameter W`

        $\mathbf{W}^{(t)} \leftarrow \mathbf{W}^{(t-1)} - \eta \nabla_{\mathbf{W}} \mathsf{Loss}^1(F^{(t-1)})$ ;

        $\mathbf{Q}^{(t)} \equiv \mathbf{Q}^{(t-1)}$;

**Stage 2:** Learning two-step reasoning for length extension ($\mathcal{T}^2$) ;

    **for** $t = T_1 + 1$ **to** $T_2$ **do**              `// Update the attention parameter Q`

        $\mathbf{Q}^{(t)} \leftarrow \mathbf{Q}^{(t-1)} - \eta \nabla_{\mathbf{Q}} \sum_{\ell=1}^2 \mathsf{Loss}_5^{2,\ell}(F^{(t-1)})$ ;

        $\mathbf{W}^{(t)} \equiv \mathbf{W}^{(t-1)}$;

**Output:** Model $F^{(T_1+T_2)}$.

---

---

**Algorithm 2:** Recursive self-training for symmetry actions

---

**Input:** Model $F^{(0)}$ with parameters $(\mathbf{W}^{(0)}, \mathbf{Q}^{(0)})$; Learning rate $\eta$; Error degree $\mathsf{E}_1 > 0$
       (constant) ; $\tau_1, \tau_2 = \widetilde{O}(\frac{\mathsf{poly}d}{\eta})$; Total Stage $K$.

**Stage 1.1:** Train FFN for one-step reasoning ($\mathcal{T}^1$) ;

    **for** $t = 1$ **to** $\tau_1$ **do**                     `// Update the FFN parameter W`

        $\mathbf{W}^{(t)} \leftarrow \mathbf{W}^{(t-1)} - \eta \nabla_{\mathbf{W}} \mathsf{Loss}^1(F^{(t-1)})$ ;

        $\mathbf{Q}^{(t)} \equiv \mathbf{Q}^{(t-1)}$;

**Stage 1.2:** Train attention for length extension ($\mathcal{T}^2$) ;

    **for** $t = \tau_1 + 1$ **to** $\tau_1 + \tau_2$ **do**        `// Update the attention parameter Q`

        $\mathbf{Q}^{(t)} \leftarrow \mathbf{Q}^{(t-1)} - \eta \nabla_{\mathbf{Q}} \mathsf{Loss}_5^{2,2}(F^{(t-1)})$ ;

        $\mathbf{W}^{(t)} \equiv \mathbf{W}^{(t-1)}$;

    $T_1 \leftarrow t$;

**Till Stage $K$:** Recursive self-train for length extension ;

    **for** $k = 2$ **to** $K$ **do**                     `// Stage k to solve `$\mathcal{T}^{2^k}$

        $L \leftarrow 2^k, \widetilde{F}^{(k)} \leftarrow F^{(T_{k-1})}$;

        **while** $\mathsf{Loss}_{\widetilde{F}^{(k)},5}^{L,2}(F^{(t-1)}) > \frac{1}{d^{\mathsf{E}_1}}$ **do**     `// Update the attention parameter Q`

            $t \leftarrow t + 1$;

            $\mathbf{Q}^{(t)} \leftarrow \mathbf{Q}^{(t-1)} - \eta \nabla_{\mathbf{Q}} \mathsf{Loss}_{\widetilde{F}^{(k)},5}^{L,2}(F^{(t-1)})$;

            $\mathbf{W}^{(t)} \equiv \mathbf{W}^{(t-1)}$;

        $T_k \leftarrow t$;

**Output:** Models $\{F^{(T_k)}\}_{k=1}^K$.

---

the problem. We expect the insight from our analysis to extend to richer tasks and inspire practical mitigation strategies, e.g., *context engineering* [99].

## C  Proof Overview

In this section, we outline the proof ideas for the main theorem. Our training schemes in Algorithms 1 and 2 alternate between two phases. First, we train the FFN parameters $\mathbf{W}$ to solve the one-step task $\mathcal{T}^1$. Then, holding $\mathbf{W}$ fixed, we train the attention parameters $\mathbf{Q}$ to solve $\mathcal{T}^2$ and, for symmetry group actions, recursively $\mathcal{T}^{2^k}$. This mirrors the division of labor in our setting (Figure 3): the FFN learns the *local update rule*, while the attention layer learns to *route and compose* these updates by locating relevant context over long sequences.

Guided by this picture, our proof overview proceeds in two parts: (1) learning the one-step mechanism for LEGO ($\mathcal{T}^1$): in-context variable retrieval (Section C.1.1) and group operations (Section C.1.2). (2) learning the attention layer: direct short-to-long generalization on $\mathcal{T}^2$ under simply transitive actions (Section C.2.1), and recursive length generalization via self-training on $\mathcal{T}^{2^k}$ under symmetry group actions (Section C.2.2).

**Notations** Let us first define a few notations to facilitate the presentation of the proofs. For each $i \in [5]$, $j \in [d]$, $r \in [m]$, define

$$\Lambda_{i,j,r}\left(\mathbf{Z}^{L,\ell-1}\right) \triangleq \sum_{\mathbf{k}\in\mathcal{I}^{L,\ell-1}} \mathbf{Attn}_{\mathsf{ans},\ell-1\to\mathbf{k}} \cdot \left\langle \mathbf{W}_{i,j,r}, \mathbf{Z}_{\mathbf{k}} \right\rangle + b_{i,j,r}. \tag{13}$$

The quantity $\Lambda_{i,j,r}$ is the FFN pre-activation, i.e., the input to $\mathbf{sReLU}$, for token position $i$, vocabulary index $j$, and hidden unit $r$. According to (3), given $\mathbf{Z}^{L,\ell-1}$, the model's output at token position $i$ and vocabulary index $j$ is

$$\left[F_i\left(\mathbf{Z}^{L,\ell-1}\right)\right]_j = \sum_{r\in[m]} \mathbf{sReLU}\big(\Lambda_{i,j,r}(\mathbf{Z}^{L,\ell-1})\big). \tag{14}$$

We denote $\mathbf{W}_i \triangleq \{\mathbf{W}_{i,j,r}\}_{j\in[d],,r\in[m]}$ as the FFN parameters associated with token position $i$. Each $\mathbf{W}_{i,j,r} \in \mathbb{R}^{5d}$ is written as a vertical concatenation $\mathbf{W}_{i,j,r} = [\mathbf{W}_{i,j,r,1}; \dots; \mathbf{W}_{i,j,r,5}]$, where $\mathbf{W}_{i,j,r,i'} \in \mathbb{R}^d$ for $i' \in [5]$; these five blocks align with the five token-type inputs.

**Additional Technical Assumptions.** Additionally, we introduce a couple of technical assumptions to be used throughout the proof. To control rare large deviations in the logits during training-time analysis, we adopt a bounded-output assumption as stated below.

**Assumption C.1** (Logit clipping). There exists $B = C_B \log d$, for some sufficiently large constant $C_B > 0$, such that each coordinate of the raw model output $F_i$ is clipped from above:

$$[F_i]_j \leftarrow \min\{[F_i]_j, B\} \qquad \text{for all } i, j.$$

This coordinate-wise clipping is a technical device to control large-deviation tails and simplify the analysis of the dynamics; $B$ can be chosen large enough to avoid interfering with the regimes we study.

Moreover, to simplify the analysis of attention dynamics, we impose a fixed block-sparsity pattern on the attention parameter $\mathbf{Q}$.

**Assumption C.2** (Block-sparse attention matrix). Let $\mathbf{Q} = [\mathbf{Q}_{p,q}]_{p,q\in[5]} \in \mathbb{R}^{5d\times 5d}$ be partitioned into $5 \times 5$ blocks with $\mathbf{Q}_{p,q} \in \mathbb{R}^{d\times d}$. We assume that

$$\mathbf{Q}_{p,q} \equiv \mathbf{0}_{d\times d} \quad \text{for all } (p,q) \notin \{(4,3),(4,4)\},$$

i.e., only the blocks $(4,3)$ and $(4,4)$ are trainable.

This block-sparsity pattern, which zeros out most inter-token attention, is standard in recent theoretical analyses of transformer training dynamics [83, 84, 85, 100, 101]. Importantly, although sparse at the $5 \times 5$ token level, the two retained blocks $(4,3)$ and $(4,4)$ are fully *dense* $d \times d$ matrices trained without constraints, leaving $2d^2$ free parameters and thus a substantive, non-trivial learning problem.

## C.1 Learning One-Step Reasoning

For solving task $\mathcal{T}^1$, in stage 1 (stage 1.1 for symmetry group task), we train $\mathbf{W}$ via the full loss $\mathsf{Loss}^1$, which contains the prediction loss across five tokens in the answer clause and $\mathbf{W}_i$ is responsible for predicting the $i$-th token in the answer clause. Thus, there are three different types of prediction tasks here: the $\langle\mathsf{blank}\rangle$ tokens (tokens 1, 2, 3), the correct variable $x_1$ (token 4) and the action update $y_1 = g_1(y_0)$ (token 5). Notice that the $\langle\mathsf{blank}\rangle$ tokens are deterministic, the learning task is straightforward. We therefore focus on the learning dynamics for the 4th and 5th tokens, which involve in-context retrieval ($\mathbf{W}_4$) and one-step group action ($\mathbf{W}_5$). Across these stage, the attention layer is fixed and keeps as uniform attention due to the zero initialization of the attention matrix, i.e., the input for the FFN layer is $\frac{1}{2}\mathbf{Z}_{\mathsf{pred},1} + \frac{1}{2}\mathbf{Z}_{\mathsf{ans},0}$.

### C.1.1 Learning In-Context Retrieval of Variables

Given the input $\frac{1}{2}\mathbf{Z}_{\mathsf{pred},1} + \frac{1}{2}\mathbf{Z}_{\mathsf{ans},0}$ for the FFN layer, the goal is to predict the fourth token in the answer clause $\mathbf{Z}_{\mathsf{ans},1}$, which is the variable $x_1$. Intuitively, the network should retrieve the occurrence

of $x_1$ in the first predicate token $\mathbf{Z}_{\mathsf{pred},1}$ and copy it to the target position. Specifically, the FFN pre-activation for predicting the fourth token to be $j$ is

$$\Lambda_{4,j,r}\left(\mathbf{Z}^{1,0}\right) = \tfrac{1}{2}\left\langle \mathbf{W}_{4,j,r}, \mathbf{Z}_{\mathsf{pred},1}\right\rangle + \tfrac{1}{2}\left\langle \mathbf{W}_{4,j,r}, \mathbf{Z}_{\mathsf{ans},0}\right\rangle + b_{4,j,r}.$$

Using the 5-vector decomposition of $\mathbf{W}_{4,j,r}$, and the fact that the embedding of $\langle\mathsf{blank}\rangle$ tokens are zero vectors (recall $e_{\tau(\cdot)}$ denote the token embedding vectors), we further obtain:

$$\Lambda_{4,j,r}\left(\mathbf{Z}^{1,0}\right) = \tfrac{1}{2}\Big(\langle \mathbf{W}_{4,j,r,1}, e_{\tau(x_1)}\rangle + \langle \mathbf{W}_{4,j,r,2}, e_{\tau(g_1)}\rangle + \langle \mathbf{W}_{4,j,r,3}, e_{\tau(x_0)}\rangle\Big) \qquad (15)$$
$$+ \tfrac{1}{2}\Big(\langle \mathbf{W}_{4,j,r,4}, e_{\tau(x_0)}\rangle + \langle \mathbf{W}_{4,j,r,5}, e_{\tau(y_0)}\rangle\Big) + b_{4,j,r}.$$

Therefore, the main idea of our analysis is to track the training dynamics of $\langle \mathbf{W}_{4,j,r,p}, e_{s'}\rangle$ for $j, s' \in [d]$, $p \in [5]$, and $r \in [m]$. Letting $s \in \tau(\mathcal{X})$ be an embedding index for a variable, our analysis shows the diagonal correlations $\langle \mathbf{W}_{4,s,r,1}, e_s\rangle$ receive strictly larger updates than all other $\langle \mathbf{W}_{4,j,r,p}, e_{s'}\rangle$. This occurs because $x_1$ co-occurs simultaneously as the first input token at $\mathbf{Z}_{\mathsf{pred},1}$ and as the supervised target at $\mathbf{Z}_{\mathsf{ans},1}$, which amplifies the gradient on $\langle \mathbf{W}_{4,s,r,1}, e_s\rangle$ when $s = \tau(x_1)$. In contrast, non-target coordinates (off-diagonals, wrong variables, value tokens, and group-action tokens) incur negligible gradients and remain $o(1)$ throughout training. Hence the correct variable's diagonal signal becomes order-wise larger, and the *active diagonal mass* $\sum_r \langle \mathbf{W}_{4,s,r,1}, e_s\rangle$ dominately grows until the end of training.

Given the dominance above, the learned weights align so that $\mathbf{W}_{4,s,r,1}$ points toward $e_s$ for $s \in \tau(\mathcal{X})$. Substituting this alignment into (15) shows that, when $j = \tau(x_1)$, the red term $\langle \mathbf{W}_{4,\tau(x_1),r,1}, e_{\tau(x_1)}\rangle$ contributes dominantly to $\Lambda_{4,\tau(x_1),r}(\mathbf{Z}^{1,0})$, thereby realizing the intended in-context retrieval: the model copies $x_1$ from the frist position of $\mathbf{Z}_{\mathsf{pred},1}$ to the fourth position of $\mathbf{Z}_{\mathsf{ans},1}$.

### C.1.2 Learning the Group Actions

For task $\mathcal{T}^1$, the FFN input $\tfrac{1}{2}\mathbf{Z}_{\mathsf{pred},1} + \tfrac{1}{2}\mathbf{Z}_{\mathsf{ans},0}$ already contains the current value $y_0$ and the action $g_1$, with no distracting information. Accordingly, when predicting the next value $y_1$, the role of $\mathbf{W}_5$ is to correctly apply the action $g_1$ to the current value $y_0$. In this section, we sketch how the model learns to implement simply transitive group actions, and we briefly discuss the symmetry case.

We first introduce notation to explain what the model should learn.

**Definition C.1** (Combinations, simply transitive actions). Assuming the group $\mathcal{G}$ follows Assumption 4.1, for each class index $j \in \tau(\mathcal{Y})$, define the *combinations*

$$\Phi := \bigcup_{j \in \tau(\mathcal{Y})} \Phi_j^\star, \quad \text{where } \Phi_j^\star := \{(g', y') \in \mathcal{G} \times \mathcal{Y} : \tau(g'(y')) = j\}.$$

$|\Phi| = n_y^2$ for the simply transitive case. We call $\phi = (g, y) \in \Phi_j^\star$ a *combination* for predicting $j = \tau(g(y))$. Hence, the goal of the model is to correctly identify all of these $\phi = (g, y) \in \Phi$.

Analogously to (15), the pre-activation at the fifth token for class $j$ and neuron $r$ is

$$\Lambda_{5,j,r}(\mathbf{Z}^{1,0}) = \tfrac{1}{2}\left\langle \mathbf{W}_{5,j,r,2}, e_{\tau(g_1)}\right\rangle + \tfrac{1}{2}\left\langle \mathbf{W}_{5,j,r,5}, e_{\tau(y_0)}\right\rangle + \text{other terms},$$

where slot "2" reads the action token $g_1$ and slot "5" reads the current value token $y_0$. We then define the following feature-magnitude notation:

**Definition C.2** (V-Notations). Given $\phi = (g, y) \in \Phi$, for the $r$-th neuron in the $j$-th class with $j \in \tau(\mathcal{Y})$ and $r \in [m]$, i.e., $\mathbf{W}_{5,j,r} \in \mathbb{R}^{5d}$, we define

$$V_{j,r}(g) := \langle \mathbf{W}_{5,j,r,2}, e_g\rangle, \quad V_{j,r}(y) := \langle \mathbf{W}_{5,j,r,5}, e_y\rangle,$$

and the composite feature magnitude

$$V_{j,r}(\phi) := \tfrac{1}{2}\big(V_{j,r}(g) + V_{j,r}(y)\big).$$

Then $V_{j,r}(\phi)$ is exactly the contribution of the input pair $(g, y)$ to $\Lambda_{5,j,r}(\mathbf{Z}^{1,0})$ for predicting $j$. Notice that for each class $j$, there are $m$ associated neurons $\{\mathbf{W}_{5,j,r}\}_{r=1}^m$, and we index neurons by the pair $(j, r)$ to emphasize that neuron $r$ is specific to class $j$; an index $r$ alone has no cross-class meaning in this context.

The main proof idea is to track $V_{j,r}(\phi)$ throughout training and show that it amplifies the correct correlations across the combinations $\Phi$ while suppressing the incorrect ones. To make this concrete

and to clarify the roles of different neurons, we introduce the following neuron–feature index set:

$$\Psi := \{(j, r, \phi) \mid j \in \tau(\mathcal{Y}), \ r \in [m], \ \phi \in \Phi\}.$$

Here, $(j, r)$ again refers to neuron $r$ for class $j$, as in Definition C.2. With this notation we can write $V_{j,r}(\phi)$ as $V_\psi$ for $\psi = (j, r, \phi) \in \Psi$.

Our proof shows that, given $j$, for each $\phi \in \Phi_j^\star$, there is exactly one neuron $r$ in $\mathbf{W}_{5,j,r}$ that is activated, denoted $r_{g \cdot y}$, to learn $\phi$; that is, $V_{j,r_{g \cdot y}}(\phi)$ will grow to a large value. Therefore, in total $n_y^2$ distinct neurons will be activated to learn all combinations, i.e., $\{(j, r_{g \cdot y}, (g, y)), \forall j \in \tau(\mathcal{Y}), (g, y) \in \Phi_j^\star\}$. The magnitude of remaining $V_\psi$ with non-activated neurons will stay close to the initialization. Our analysis shows that learning follows an *implicit curriculum* induced by the magnitude of features $V_\psi$ at initialization:

$$\psi \prec \psi' \iff V_\psi^{(0)} \geq V_{\psi'}^{(0)}, \quad \forall \psi, \psi' \in \Psi.$$

Items on the left under this ordering are learned first, and those on the right are learned later. We denote by $\Sigma^\star$ the *learning curriculum*: the ordered set of neuron–feature indices $\{(j, r_{g \cdot y}, (g, y))\}$ identified above, equipped with this order.

Then, for each $\psi \in \Sigma^\star$, the learning process follows its associated ordering, and when it is the turn of $\psi = (j, r_{g \cdot y}, (g, y))$, the learning process is mainly characterized by the following two phases:

- **Phase I:** Emergence of the feature $V_\psi$ among other features. During this phase, $V_\psi$ grows faster than any $\psi' \neq \psi$ with $\psi \prec \psi'$, while not affecting the already learned predecessors $\psi' \in \Sigma^\star$ with $\psi' \prec \psi$, with growth rate

$$V_\psi^{(t+1)} \geq V_\psi^{(t)} + \widetilde{\Omega}(\eta) \cdot \left(b_{j,r} + V_\psi^{(t)} \pm o(\mu)\right)^{q-1}. \tag{16}$$

  This form permits the application of the tensor power method (TPM) [81] and explains the ordering induced by the magnitude of $V_\psi$ at initialization, since TPM implies that a slightly larger initial value leads to dramatically faster growth. By the end of Phase I, for any other feature $\psi' \neq \psi$ with $\psi \prec \psi'$, the growth of that feature is capped at $\widetilde{O}(\sigma_0)$, its initial magnitude.

- **Phase II:** Growth of $V_\psi$ and cancellation of incorrect features. After Phase I, the target feature $V_\psi$ already has a relatively large magnitude; however, features in the set $\{\psi' = (j, r_{g \cdot y}, \phi') \in \Psi \mid \phi' = (g', y) \text{ or } \phi' = (g, y')\}$, namely, the features in the same neuron $(j, r_{g \cdot y})$ that share exactly one component of $\phi = (g, y)$, may also grow as the shared component increases. Note that $j$ is not the correct label for such $\phi'$, and we call them **confounding features**. The key characterization in this phase is that, due to a stationarity property of the gradients, although the dynamics are coupled, the wrong half of the confounding features grows in the negative direction while the correct half continues to grow, and ultimately the confounding feature cancels out.

We show that for each $\psi \in \Sigma^\star$ that should be learned, it retains its structure after its own learning process (i.e., at the end of Phase II) and persists through the final convergence while the other features in $\Sigma^\star$ are learned. Specifically, we have the following properties at the end of training:

$$V_{j,r_{g \cdot y}}(g), \ V_{j,r_{g \cdot y}}(y) \approx B, \tag{17a}$$

$$V_{j,r_{g \cdot y}}(g'), \ V_{j,r_{g \cdot y}}(y') \approx -B, \tag{17b}$$

$$V_{j,r_{g \cdot y}}(g) + V_{j,r_{g \cdot y}}(y') \leq o(1) \text{ and } V_{j,r_{g \cdot y}}(g') + V_{j,r_{g \cdot y}}(y) \leq o(1) \quad \forall g' \neq g, \ y' \neq y. \tag{17c}$$

**How does this structure perform group actions?** For an input pair $(g, y)$, let the correct answer be $j = \tau(g(y))$. Then the pre-activation for predicting $j$ is around $B$ at neuron $r_{g \cdot y}$, with all other neurons for $j$ near the initial value $\widetilde{O}(\sigma_0)$. Thus the model output at $j$ is around $B$. For all other predictions $j'$, there exist $\phi_1 = (g', y)$ and $\phi_2 = (g, y')$ such that $j' = \tau(g'(y)) = \tau(g(y'))$. By cancellation of the incorrect features, we have $V_{j',r_{g' \cdot y}}(\phi_1), \ V_{j',r_{g \cdot y'}}(\phi_2) \leq o(1)$. Since $B = \Theta(\log d)$, this implies that the logit on the correct prediction $j$ is very close to 1.

**Symmetry Group Actions.** The proof strategy for symmetry group actions mirrors the simply transitive case through the emergence, refinement, and convergence phases. However, symmetry actions create richer interactions because multiple group elements can map the same $y$ to the same $j$, which requires more nuanced control of the training dynamics and leads to different learned feature structures. To illustrate this pattern, we slightly modify the definition of **combinations** used in the simply transitive case by introducing the notion of the **fiber** of a value.

**Definition C.3** (Combinations, Symmetry Actions). Assuming the group $\mathcal{G}$ follows Assumption 5.1, define $\text{Fiber}_{j,y} := \{g \in \mathcal{G} \mid \tau(g(y)) = j\}$.[5] This allows us to define the combinations as

$$\Phi = \{\varphi_{j,y} \mid j \in \tau(\mathcal{Y}), \, y \in \mathcal{Y}\}, \quad \text{where} \quad \varphi_{j,y} = \text{Fiber}_{j,y} \times \{y\}.$$

Moreover, $|\varphi_{j,y}| = (n_y - 1)!$. We continue to call $\phi = (g, y) \in \mathcal{G} \times \mathcal{Y}$ a combination, whereas an element of $\Phi$ is now a subset of combinations $\varphi_{j,y}$ (there are $n_y^2$ such $\varphi_{j,y}$ in total), which includes all pairs $(g, y)$ such that $g$ sends $y$ to $j$, and which reduces to the single pair $(g, y)$ in the simply transitive case.

Based on these notions, the main difference from the simply transitive case is that previously the basic learning unit is each pair $(g, y) \in \Phi$, whereas now it is the subset $\varphi_{j,y}$ of combinations. All combinations $\phi$ within $\varphi_{j,y}$ are captured by the same and unique neuron, denoted $r_{j,y}$, in the sense that the feature magnitude $V_{j,r_{j,y}}(\phi)$ is large for any $\phi \in \varphi_{j,y}$. Accordingly, the learning curriculum $\Sigma^\star$ is now based on the initial value of the ensemble of feature magnitudes: $\frac{1}{|\varphi_{j,y}|} \sum_{\phi \in \varphi_{j,y}} V_{j,r_{j,y}}(\phi)$. Finally, at convergence, we have the following imbalance of feature magnitudes:

$$V_{j,r_{j,y}}(g) \approx 2B, \qquad V_{j,r_{j,y}}(y) \approx \frac{2B}{n_y} \quad \forall\, g \in \text{Fiber}_{j,y}; \tag{18a}$$

$$V_{j,r_{j,y}}(g') \approx \frac{2B}{n_y}, \qquad V_{j,r_{j,y}}(y') \approx -2B; \tag{18b}$$

$$V_{j,r_{j,y}}(g) + V_{j,r_{j,y}}(y') \leq o(1) \text{ and } V_{j,r_{j,y}}(g') + V_{j,r_{j,y}}(y) \leq o(1) \quad \forall\, g' \notin \text{Fiber}_{j,y}, \, y' \neq y. \tag{18c}$$

### C.1.3 What Changes for Longer Tasks?

**FFN is length invariant.** The feed-forward network (with weights $\mathbf{W}$) only acts on the *attended linear combination* at the current output clause, rather than scanning the sequence to retrieve information. Retrieval from the context is delegated to the attention layer. Hence, as sequence length grows and positions shift, the FFN computation remains the same mapping on its local input, making it length- and position-invariant. On the other hand, uniform attention becomes increasingly diluted as irrelevant clauses accumulate, so selective attention and learning attention layer are required.

**Desired attention patterns for predicting each tokens.** To predict the fourth token, the model only needs the variable $x_{\ell+1}$ from the predicate clause $\mathbf{Z}_{\text{pred},\ell+1}$. If the attention mass on $\mathbf{Z}_{\text{pred},\ell+1}$ does not vanish with $L$, the same pattern learned in $\mathcal{T}^1$ with $\mathbf{W}_4$ applies directly. Predicting the fifth token is harder because it requires *two* retrievals: (i) the group action from $\mathbf{Z}_{\text{pred},\ell+1}$, and (ii) the current value from $\mathbf{Z}_{\text{ans},\ell}$, followed by applying the update via $\mathbf{W}_5$. Thus robustness over length hinges on maintaining both attention links. If the attention layer is robust enough to support the fifth-token prediction, then the fourth-token prediction follows as a special case, since it needs only one of the two attention links to persist. Accordingly, in what follows we focus on optimizing the loss for the fifth token; achieving high accuracy there is effectively equivalent to solving the entire task.

### C.2 Learning the Attention Layer

Successful training on $\mathcal{T}^1$ shows that the model has learned the one-step update $g \circ y$. Turning to task $\mathcal{T}^2$, the remaining difficulty is *routing*: directing attention to the appropriate locations. Thus we train the attention matrix $\mathbf{Q}$ to learn the routing pattern and keep $\mathbf{W}$ fixed. In this part, we will show how training induces the **attention concentration** pattern highlighted in Theorems 4.1 and 5.1: given an input $\mathbf{Z}^{L,\ell-1}$, the attention mass concentrates on $\mathbf{Attn}_{\text{ans},\ell-1\to\text{pred},\ell}$ and $\mathbf{Attn}_{\text{ans},\ell-1\to\text{ans},\ell-1}$. We quantify routing quality via the *attention concentration degree*

$$\epsilon_{\text{attn}}^{L,\ell}\left(\mathbf{Z}^{L,\ell-1}\right) = 1 - \mathbf{Attn}_{\text{ans},\ell-1\to\text{pred},\ell}\left(\mathbf{Z}^{L,\ell-1}\right) - \mathbf{Attn}_{\text{ans},\ell-1\to\text{ans},\ell-1}\left(\mathbf{Z}^{L,\ell-1}\right), \tag{19}$$

which measures the fraction of attention mass *not* placed on the two key clauses, and the *attention gap*

$$\Delta^{L,\ell}\left(\mathbf{Z}^{L,\ell-1}\right) = \left| \mathbf{Attn}_{\text{ans},\ell-1\to\text{pred},\ell}\left(\mathbf{Z}^{L,\ell-1}\right) - \mathbf{Attn}_{\text{ans},\ell-1\to\text{ans},\ell-1}\left(\mathbf{Z}^{L,\ell-1}\right) \right|, \tag{20}$$

---

[5]  We use the notion of fiber to denote the left cosets $gG_y \subset \mathcal{G}$, $g \in \mathcal{G}$, where $y \in \mathcal{Y}$ and $G_y = \{g \in \mathcal{G} \mid g(y) = y\}$ is the stabilizer. Since the fiber of the orbit map $f_y(g) = g(y)$ is the preimage of $f_y$, i.e., $f_y^{-1}(y')$ with $y' = \tau^{-1}(j)$, $j \in \tau(\mathcal{Y})$, this is exactly the set $\text{Fiber}_{j,y}$.

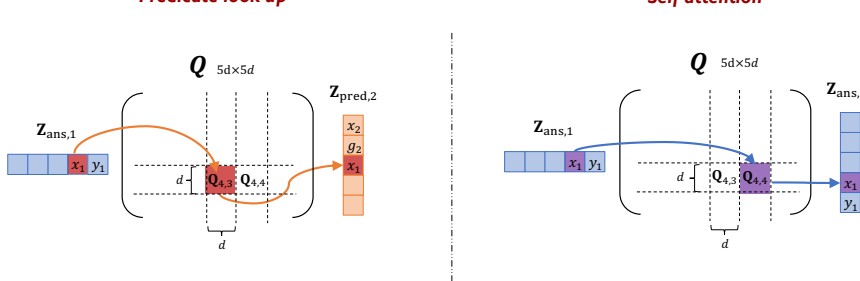

Figure 7: The illustration of how different components of the attention matrix $\mathbf{Q}$ are used to route the attention to the appropriate locations. The query clause is $\mathbf{Z}_{\text{ans},1}$ and the goal is to retrieve the correct action $g_2$ from $\mathbf{Z}_{\text{pred},2}$ and value $y_1$ from the current answer clause $\mathbf{Z}_{\text{ans},1}$. $\left[\mathbf{Q}_{4,p}\right]_{s,s}$ will grow and dominate the learning dynamics for $p \in \{3,4\}$ and $s \in \tau(\mathcal{X})$. Thus, in this example, large $\left[\mathbf{Q}_{4,3}\right]_{\tau(x_1),\tau(x_1)}$ indicates the large attention to the predicate clause $\mathbf{Z}_{\text{pred},2}$ and large $\left[\mathbf{Q}_{4,4}\right]_{\tau(x_1),\tau(x_1)}$ indicates the large self-attention to the answer clause $\mathbf{Z}_{\text{ans},1}$.

which captures how balanced the two target attentions are. Thus, effective routing corresponds to small $\epsilon_{\text{attn}}^{L,\ell}$ (high concentration) and small $\Delta^{L,\ell}$ (good balance).

Under Assumption C.2, for an input $\mathbf{Z}^{L,\ell-1}$ the (unnormalized) attention score from $\mathbf{Z}_{\text{ans},\ell-1}$ to a clause $\mathbf{Z}_{\mathbf{k}}$ decomposes as

$$\mathbf{Z}_{\text{ans},\ell-1}^{\top}\mathbf{Q}\mathbf{Z}_{\mathbf{k}} \; = \; \mathbf{Z}_{\text{ans},\ell-1,4}^{\top}\mathbf{Q}_{4,3}\mathbf{Z}_{\mathbf{k},3} \; + \; \mathbf{Z}_{\text{ans},\ell-1,4}^{\top}\mathbf{Q}_{4,4}\mathbf{Z}_{\mathbf{k},4},$$

where $\mathbf{Z}_{\mathbf{k}} = [\mathbf{Z}_{\mathbf{k},1}, \ldots, \mathbf{Z}_{\mathbf{k},5}]$ with $\mathbf{Z}_{\mathbf{k},i} \in \mathbb{R}^d$. By design, in the clause embeddings the *fourth* token of a *predicate* clause and the *third* token of an *answer* clause are $\langle\text{blank}\rangle$. Consequently, we further have

$$\mathbf{Z}_{\text{ans},\ell-1}^{\top}\mathbf{Q}\mathbf{Z}_{\text{pred},\ell'} = \mathbf{Z}_{\text{ans},\ell-1,4}^{\top}\mathbf{Q}_{4,3}\mathbf{Z}_{\text{pred},\ell',3}, \qquad \mathbf{Z}_{\text{ans},\ell-1}^{\top}\mathbf{Q}\mathbf{Z}_{\text{ans},\ell'} = \mathbf{Z}_{\text{ans},\ell-1,4}^{\top}\mathbf{Q}_{4,4}\mathbf{Z}_{\text{ans},\ell',4}, \tag{21}$$

which means $\mathbf{Q}_{4,3}$ governs attention to predicate clauses, while $\mathbf{Q}_{4,4}$ governs attention to answer clauses. An immediate observation from (21) is that the desired allocation can be realized by growing the diagonal entries $[\mathbf{Q}_{4,3}]_{s,s}$ and $[\mathbf{Q}_{4,4}]_{s,s}$[6] for $s \in \tau(\mathcal{X})$ since $x_{\ell-1}$ appears as the third token in $\mathbf{Z}_{\text{pred},\ell}$ and as the fourth token in $\mathbf{Z}_{\text{ans},\ell-1}$ simultaneously. Our analysis of tracking the attention dynamics will show that these diagonal coordinates indeed receive asymptotically larger gradient magnitudes than all other entries due to strong co-occurrence signal, and the training dynamics are dominated by their growth. See Figure 7 for an illustration.

For notational simplicity, we will refer to the relevant diagonal entries $[\mathbf{Q}_{4,3}]_{s,s}$ and $[\mathbf{Q}_{4,4}]_{s,s}$ simply as $\mathbf{Q}_{4,3}$ and $\mathbf{Q}_{4,4}$ below. The remainder of the proof quantifies how the growth of $\mathbf{Q}_{4,3}$ and $\mathbf{Q}_{4,4}$ simultaneously drives the concentration degree $\epsilon_{\text{attn}}^{L,\ell}$ toward zero and controls the gap $\Delta^{L,\ell}$, ensuring that the FFN consistently receives $(g_\ell, y_{\ell-1})$ and thus outputs the correct $y_\ell$.

### C.2.1 Simply Transitive Actions

For the simply transitive case, we analyze the gradient contribution at position $i = 5$ on task $\mathcal{T}^2$, i.e., the loss $\sum_{\ell=1}^{2} \text{Loss}_5^{2,\ell}$. We show that for $\mathcal{T}^2$ the *attention concentration degree* $\epsilon_{\text{attn}}^{2,\ell}$ (for $\ell \in [2]$) can be reduced below $O(1/d^{c^*})$ for some constant $0 < c^* < 1$, indicating highly focused mass on the relevant clauses. When irrelevant entries of $\mathbf{Q}$ are small, we also have $\epsilon_{\text{attn}}^{2,1} \leq \epsilon_{\text{attn}}^{2,2}$, since the number of irrelevant clauses doubles from $\ell = 1$ to $\ell = 2$; hence we focus on controlling $\epsilon_{\text{attn}}^{2,2}$.

- **Stage 2.1: Growth of an initial gap.** Early in training, attention is close to uniform, so given $\mathbf{Z}^{2,\ell-1}$ we have the approximations

$$\Lambda_{5,j,r}(\mathbf{Z}^{2,0}) \; \approx \; \tfrac{1}{3}\, V_{j,r}(g_1) + \tfrac{1}{3}\, V_{j,r}(g_2) + \tfrac{1}{3}\, V_{j,r}(y_0)\,,$$

---

[6] $\quad [\mathbf{A}]_{s,s'}$ denotes the entry of the matrix $\mathbf{A}$ at the $s$-th row and $s'$-th column

$$\Lambda_{5,j,r}(\mathbf{Z}^{2,1}) \;\approx\; \tfrac{1}{4}\,V_{j,r}(g_1) + \tfrac{1}{4}\,V_{j,r}(g_2) + \tfrac{1}{4}\,V_{j,r}(y_0) + \tfrac{1}{4}\,V_{j,r}(y_1)\;.$$

By the cancellation at the convergence in (17), for $\ell = 2$ all $\Lambda$ lie in the small smoothed regime, whereas for $\ell = 1$ we obtain a correct logit for $y_1 = g_1(y_0)$ and a spurious logit for $g_2(y_0)$ of magnitude about $B/3$. Consequently, $-\nabla_{\mathbf{Q}}\mathsf{Loss}_5^{2,2}$ is negligible, while $-\nabla_{\mathbf{Q}}\mathsf{Loss}_5^{2,1}$ is comparatively large and drives $\mathbf{Q}_{4,3}$ to grow faster than $\mathbf{Q}_{4,4}$ (increasing $\mathbf{Q}_{4,4}$ would also amplify the wrong prediction $\tau(g_2(y_0))$ and thus not reduce $\mathsf{Loss}_5^{2,1}$). An $\Omega(1/\log d)$ gap emerges between the diagonals $[\mathbf{Q}_{4,3}]_{s,s}$ and $[\mathbf{Q}_{4,4}]_{s,s}$, yielding an early routing advantage toward right predicate clause.

- **Stage 2.2: Joint growth with a controlled gap.** As $\mathbf{Q}_{4,3}$ increases, the weight $\mathbf{Attn}_{\mathsf{ans},1\to\mathsf{pred},2}$ becomes large, moving $\Lambda_{5,\tau(g_2(y_1)),r_{g_2\cdot y_1}}$ and $\Lambda_{5,\tau(g_2(y_0)),r_{g_2\cdot y_0}}$ for $\ell = 2$ into the linear regime. Gradients from $\ell = 2$ then dominate, and $\mathbf{Q}_{4,4}$ starts to grow to separate the correct $y_2 = g_2(y_1)$ from the incorrect $\tau(g_2(y_0))$. Throughout, the gap between $\mathbf{Q}_{4,3}$ and $\mathbf{Q}_{4,4}$ stays within $[\Omega(1/\log d),\,O(1)]$, so the attention gap satisfies $\Delta^{2,2} = \Omega(1/\log d)$.

- **Stage 2.3: Convergence and gap reduction.** Continued joint growth of $\mathbf{Q}_{4,3}$ and $\mathbf{Q}_{4,4}$ concentrates attention near its ideal limit, making $\epsilon_{\mathsf{attn}}^{2,2}$ small. We show $\Delta^{2,2}$ cannot remain above $o(1)$ for long; otherwise an incorrect logit $\mathbf{logit}_{5,\tau(g_2(y_0))}$ would acquire a stronger gradient and force $\mathbf{Q}_{4,4}$ to outpace $\mathbf{Q}_{4,3}$, which contradicts stability. Therefore, at convergence we have (i) $\epsilon_{\mathsf{attn}}^{2,2} = O(d^{-c^*})$ for some $c^* \in (0,1)$ and (ii) $\Delta^{2,2} = o(1)$; hence both $\mathbf{Q}_{4,3}$ and $\mathbf{Q}_{4,4}$ equal $C\log d \pm o(1)$, while all other entries of the attention matrix remain close to their initial values, where $C > 0$ depends on $c^*$.

**Strong length generalization.** The key is that attention concentrates cleanly while remaining stably balanced. For $\mathcal{T}^L$, we obtain

$$\epsilon^{L,\ell} \;\leq\; \frac{O(1)\cdot L}{O(1)\cdot L + 1/\epsilon_{\mathsf{attn}}^{2,2}} \;\leq\; \frac{1}{1 + \Omega(d^{c^*}/L)}.$$

Hence the model tolerates $O(d^{c^*})$ irrelevant clauses: in particular, if $L = o(d^{c^*})$ then $\epsilon^{L,\ell} = o(1)$; and if $L = \Theta(d^{c^*})$ then $\epsilon^{L,\ell} \leq \frac{1}{1+\Omega(1)}$, which can be made small by choosing the proportionality constant in $L = \Theta(d^{c^*})$ appropriately. Moreover, $\Delta^{L,\ell} \leq \Delta^{2,2} \leq o(1)$, so the two target attention masses remain balanced as $L$ grows, preventing errors from large imbalance (e.g., predicting $\tau(g_2(y_0))$ in Stage 2.3). Consequently, the correct logit satisfies $1 - \mathbf{logit}_{5,\tau(g_{\ell+1}(y_\ell))} \leq \frac{1}{\mathsf{poly}(d)}$, so $\mathcal{T}^L$ is solved with accuracy $1 - 1/\mathsf{poly}(d)$.

### C.2.2 Symmetry Group Actions

We now turn to symmetry group tasks $\mathcal{T}^L$ and analyze GD updates with respect to the per-token loss $\mathsf{Loss}_5^{L,2}$ (i.e., predicting the value token in $\mathbf{Z}_{\mathsf{ans},2}$ from $\mathbf{Z}^{L,1}$).

**The case $L = 2$.** The high-level picture mirrors the simply transitive case, but because multiple group elements can map a given $y$ to the same $j$, the learned $V_{j,r}(g)$ and $V_{j,r}(y)$ structures are unbalanced. Moreover, as we discussed in the hardness part for the symmetry group in Section 5, there will be a non-negligible proportions of distractor clauses. These make it harder to keep a tight balance between $\mathbf{Q}_{4,3}$ and $\mathbf{Q}_{4,4}$ as in the simply transitive case. Nevertheless, we prove that both $\mathbf{Q}_{4,3}$ and $\mathbf{Q}_{4,4}$ grow, and the attention gap $\Delta^{2,2}$ is controlled by a feedback mechanism: if $\Delta^{2,2}$ exceeds a small fixed threshold (in either direction), some incorrect logit receives a stronger gradient, which pushes the system back toward balance. Consequently, after sufficient training: (i) $\epsilon_{\mathsf{attn}}^{2,2} \leq C_1$; (ii) $\Delta^{2,2} \leq C_2$, for sufficiently small constants $C_1, C_2$; and since $B = \Theta(\log d)$, we still obtain $\mathsf{Loss}_5^{2,2} \leq 1/\mathsf{poly}(d)$.

**Recursive learning for $\mathcal{T}^{2^k}$, $k \geq 2$.** Because $\epsilon_{\mathsf{attn}}^{2^{k-1},2}$ is already a small constant, initialization for $\mathcal{T}^{2^k}$ satisfies $\epsilon_{\mathsf{attn}}^{2^k,2} \leq 2\,\epsilon_{\mathsf{attn}}^{2^{k-1},2}$ (still small), and $\Delta^{2^k,2} \leq \Delta^{2^{k-1},2}$. Thus the attention pattern remains close to that in $\mathcal{T}^{2^{k-1}}$, and $\mathbf{Z}^{2^k,1}$ follows a bootstrapped LEGO distribution generated by the greedy model $\widehat{p}_{F^{(T_{k-1})}}$, which coincides with the original LEGO source. In particular, $y_1 = g_1(y_0)$ and $y_2 = g_2(y_1)$ are correct. We can therefore reuse the convergence analysis from $\mathcal{T}^{2^{k-1}}$ to show that both $\epsilon_{\mathsf{attn}}^{2^k,2}$ and $\Delta^{2^k,2}$ decrease to a small constant, yielding stable, inductive concentration across recursive reasoning depths.

### C.3 Significance of the Proof

Our proof techniques are inspired by recent advances in understanding the dynamics of feature learning in neural networks [102, 103, 104, 105, 82, 106, 83, 84], which show how gradient-based training induces useful internal patterns and representations. Building on these ideas, we analyze the dynamics of the FFN and attention layers to capture how transformers gradually acquire length-generalizable reasoning through CoT training. We conclude this section by summarizing the technical significance of our analysis for learning in the FFN and attention layers.

**Learning the group actions.** Most existing optimization-based analyses of CoT [36, 35, 34, 37] hard-code the group action, e.g., parity, as a fixed FFN or an almost-linear map. In contrast, we *train* a nonlinear FFN end-to-end to perform the group action at each step, with parity as a special $n = 2$ instance of our simply transitive framework. This formulation is strictly more general and substantially more challenging than prior setups. Our proof shows that the model learns not only the basic action for simply transitive groups but also more complex actions for richer actions that is transitive but not free. Especially, we precisely characterizing how task-relevant features emerge and spurious features are suppressed during the process. The technique we used in the proof for learning the FFN on discrete combinations of data is of independent interest beyond analyzing CoT.

**Learning the attention patterns.** Prior work on transformer training dynamics has established the *attention concentration* principle as a key mechanism for solving various tasks, e.g., in-context learning [83], self-supervised learning [84], and graph learning [107]. In those settings, only a single token needs to be retrieved, so the attention matrix is learned to be diagonal (after a suitable change of basis), and a single pattern suffices. In contrast, our task requires retrieving two different types of clauses, so the model must learn two distinct components (blocks) in the attention matrix, $\mathbf{Q}_{4,3}$ and $\mathbf{Q}_{4,4}$. As we show in Appendix C.2, beyond growing these blocks, we must maintain a delicate balance between them. This balance is crucial for length generalization across different group actions, introduces substantial technical challenges, and our proof provides a fine-grained control of it.

## D  Learning In-Context Retrieval of Variables

In this section, we focus on the learning process of $\mathbf{W}_4$ for the task $\mathcal{T}^1$. For this task, $\mathbf{W}_4$ should predict the 4-th token in the answer clause $Z_{\mathsf{ans},1}$, which is the variable $x_1$. We will show that, the network learns to retrieve the target variable $x_1$ from the first token of the first predicate clause $Z_{\mathsf{pred},1}$ to make an accurate prediction. Throughout the rest of the proof, we will omit the subscript for the expectation $\mathbb{E}$, when the context is clear.

### D.1  Preliminaries

First we define some notations for the presentation of gradients.

**Notations for gradient expressions** For each $i \in [5], \ell \in [L], j \in [d], r \in [m]$, we denote

$$\mathcal{E}_{i,j}(\mathbf{Z}^{L,\ell-1}) \triangleq \mathbb{1}_{\mathbf{Z}_{\mathsf{ans},\ell,i}=e_j} - \mathbf{logit}_{i,j}(F, \mathbf{Z}^{L,\ell-1}), \tag{22}$$

$$\Lambda_{i,j,r}(\mathbf{Z}^{L,\ell-1}) \triangleq \sum_{\mathbf{k} \in \mathcal{I}^{L,\ell-1}} \mathbf{Attn}_{\mathsf{ans},\ell-1 \to \mathbf{k}} \cdot \langle \mathbf{W}_{i,j,r}, \mathbf{Z}_{\mathbf{k}} \rangle + b_{i,j,r}. \tag{23}$$

where $\mathbf{logit}_{i,j}(F, \mathbf{Z}^{L,\ell-1})$ are defined as

$$\mathbf{logit}_{i,j}(F, \mathbf{Z}^{L,\ell-1}) := \frac{e^{F_{i,j}(\mathbf{Z}^{L,\ell-1})}}{\sum_{j' \in [d]} e^{F_{i,j'}(\mathbf{Z}^{L,\ell-1})}}.$$

**Fact D.1.** For any $i \in [5], j \in [d], r \in [m]$, we have the following gradient expression:

$$-\nabla_{\mathbf{W}_{i,j,r}} \mathsf{Loss}^L = \mathbb{E}\Big[ \sum_{\ell=1}^{L} \mathcal{E}_{i,j}(\mathbf{Z}^{L,\ell-1}) \mathbf{sReLU}'(\Lambda_{i,j,r}(\mathbf{Z}^{L,\ell-1})) \sum_{\mathbf{k} \in \mathcal{I}^{L,\ell-1}} \mathbf{Attn}_{\mathsf{ans},\ell-1 \to \mathbf{k}} \mathbf{Z}_{\mathbf{k}} \Big].$$

For simplicity of notation, we will henceforth denote $\Lambda_{i,j,r}(\mathbf{Z}^{L,\ell-1})$ by $\Lambda_{i,j,r}$ and $\mathcal{E}_{i,j}(\mathbf{Z}^{L,\ell-1})$ by $\mathcal{E}_{i,j}$ when the context is clear.

Given $\mathbf{Z}^L$, we use $\widehat{\mathcal{X}}^L$ to denote the appeared variables in the context clauses, i.e. $\widehat{\mathcal{X}}^L = \{x_0, x_1, \ldots, x_L\}$. We write $\widehat{\mathcal{X}}^L$ as $\widehat{\mathcal{X}}$ for simplicity. Throughout this section, we write $[F_i]_j$ as $F_{i,j}$ for simplicity.

## D.2 Induction Hypothesis

**Proof Plan.** The main idea is to track the dynamics of different types of weights $\mathbf{W}_{4,j,r,p}$. Specifically, we prove that for each $j \in \tau(\mathcal{X})$, there exists certain neurons $r \in [m]$ such that the corresponding weights $\mathbf{W}_{4,j,r,1}$ grow significantly along the direction $e_j$, while all others remain small. Specifically, we proceed in four steps:

1. For $j \in \tau(\mathcal{X})$, define

$$\Gamma_{4,j}^{(t)} \triangleq \max_{r \in [m]} \langle \mathbf{W}_{4,j,r,1}^{(t)}, e_j \rangle + \sigma_0 \log d,$$

   to track the maximal activation associated with retrieving the correct variable token.

2. **Establish rapid growth (early phase).** Let $\Lambda^- = \Theta(1/m)$, and define the hitting time

$$T_{1,j} \triangleq \min\{t > 0 : \Gamma_{4,j}^{(t)} \geq \Lambda^-\}.$$

   We show that for iterations $t \leq T_1 = \Theta(d\sigma_0^{q-2}/\eta)$, the diagonal weights $\langle \mathbf{W}_{4,j,r,1}^{(t)}, e_j \rangle$ grow rapidly, causing $\Gamma_{4,j}^{(t)}$ to enter a linear growth regime. Simultaneously, the model confidently identifies the correct variable, indicated by $1 - \mathbf{logit}_{4,\tau(x_1)}^{(t)} = 1 - o(1)$.

3. **Convergence via dominant neurons (late phase).** For each $j \in \tau(\mathcal{X})$, define active neuron sets and their total activation as:

$$\mathcal{A}_{4,j}^{(t)} \triangleq \{r \in [m] : \langle \mathbf{W}_{4,j,r,1}^{(t)}, e_j \rangle \geq \varrho \log d\}, \quad \Phi_{4,j}^{(t)} \triangleq \sum_{r \in \mathcal{A}_{4,j}^{(t)}} \langle \mathbf{W}_{4,j,r,1}^{(t)}, e_j \rangle.$$

   For iterations $t > T_1$, we analyze the refined dynamics, proving that the total diagonal activation $\Phi_{4,j^\star}^{(t)}$, for the weakest activated variable $j^\star$, eventually grows to $\Theta(\log d)$, ensuring successful learning.

4. **Bounding non-target correlations.** We finally show by induction that all other correlations $\langle \mathbf{W}_{4,j,r,p}^{(t)}, e_s \rangle$, including group actions, value tokens, off-diagonal tokens, and non-target variables, remain negligible throughout the training process.

Our proof begins by positing an induction hypothesis expected to hold throughout training. We then analyze the dynamics under this hypothesis and show that it remains valid along the entire training trajectory, establishing the claim at convergence.

**Induction D.1.** *For $t \leq T = \frac{\text{poly}d}{\eta}$, all of the following holds:*

*(a). for $j \in \tau(\mathcal{X})$, $\widetilde{\Omega}(\sigma_0) \leq \langle \mathbf{W}_{4,j,r,1}^{(t)}, e_j \rangle + \mu \leq \widetilde{O}(1)$, where $\langle \mathbf{W}_{4,j,r,1}^{(t)}, e_j \rangle$ is non-decreasing;*

*(b). for $j \in \tau(\mathcal{X})$, $g \in \mathcal{G}$*

$$\left| \langle \mathbf{W}_{4,j,r,2}^{(t)}, e_{\tau(g)} \rangle \right| \leq \widetilde{O}(\sigma_0) + O\left(\frac{1}{|\mathcal{G}|}\right) \max \left\{ \langle \mathbf{W}_{4,j,r,1}^{(t)}, e_j \rangle, \min_{r' \in \mathcal{A}_{4,j^*}^{(t)}} \langle \mathbf{W}_{4,j^*,r',1}^{(t)}, e_{j^*} \rangle \right\};$$

*(c). for $j \in \tau(\mathcal{X})$, $y \in \mathcal{Y}$*

$$\left| \langle \mathbf{W}_{4,j,r,5}^{(t)}, e_{\tau(y)} \rangle \right| \leq \widetilde{O}(\sigma_0) + O\left(\frac{1}{|\mathcal{Y}|}\right) \max \left\{ \langle \mathbf{W}_{4,j,r,1}^{(t)}, e_j \rangle, \min_{r' \in \mathcal{A}_{4,j^*}^{(t)}} \langle \mathbf{W}_{4,j^*,r',1}^{(t)}, e_{j^*} \rangle \right\};$$

*(d). else, $\left| \langle \mathbf{W}_{4,j,r,p}^{(t)}, e_{j'} \rangle \right| \leq \widetilde{O}(\sigma_0)$ for any other $j, j' \in [d]$.*

**Claim D.1.** If Induction D.1 holds at iteration $t$, then for an input $\mathbf{Z}^{1,0}$, we have

1. if $j = \tau(x_1)$,

$$\Lambda_{4,j,r}^{(t)} = \frac{1}{2} \langle \mathbf{W}_{4,j,r,2}^{(t)}, e_j \rangle + \frac{1}{2} \langle \mathbf{W}_{4,j,r,2}^{(t)}, e_{\tau(g_1)} \rangle + \frac{1}{2} \langle \mathbf{W}_{4,j,r,5}^{(t)}, e_{\tau(y_0)} \rangle + \frac{5}{2} \mu + \widetilde{O}(\sigma_0);$$

2. else if $j \in \tau(\mathcal{X} \setminus \{x_1\})$,

$$\Lambda_{4,j,r}^{(t)} = \frac{1}{2}\langle \mathbf{W}_{4,j,r,2}^{(t)}, e_{\tau(g_1)}\rangle + \frac{1}{2}\langle \mathbf{W}_{4,j,r,5}^{(t)}, e_{\tau(y_0)}\rangle + \frac{5}{2}\mu + \widetilde{O}(\sigma_0);$$

3. otherwise, $0 \le \Lambda_{4,j,r}^{(t)} \le \frac{5}{2}\mu + \widetilde{O}(\sigma_0)$.

**Claim D.2.** If Induction D.1 holds at iteration $t$, then for an input $\mathbf{Z}^{1,0}$,

1. if $j = \tau(x_1)$, $\mathbf{logit}_{4,j}^{(t)} = \frac{e^{O(\Phi_{4,j}^{(t)})}}{e^{O(\Phi_{4,j}^{(t)})}+d}$;

2. otherwise, $\mathbf{logit}_{4,j}^{(t)} = O(\frac{1}{d})\left(1 - \mathbf{logit}_{\tau(x_1)}^{(t)}\right)$.

*Proof.* If $j = \tau(x_1)$, by Induction D.1 and Claim D.1, we have

$$0 \le F_{4,j}^{(t)}(\mathbf{Z}^{1,0}) \le \sum_{r\in[m]} [\Lambda_{4,j,r}^{(t)}]^{+} \le \left(\Phi_{4,j}^{(t)} + O(\frac{\max\{\Phi_{4,j}^{(t)}, \Phi_{4,j*}^{(t)}\}}{|\mathcal{G}|})\right) + \widetilde{O}(\sigma_0) + O(m\varrho \log d)$$

$$= \left(\Phi_{4,j}^{(t)} + O(\frac{\Phi_{4,j}^{(t)}}{|\mathcal{G}|})\right) + \widetilde{O}(\sigma_0) + O(\frac{1}{\text{polylog}d}).$$

For $j \in \tau(\mathcal{X}) \ne \tau(x_1)$, $F_{4,j}^{(t)}(\mathbf{Z}^{1,0}) \le \widetilde{O}(\sigma_0) + O(\frac{\max\{\Phi_{4,j}^{(t)}, \Phi_{4,j*}^{(t)}\}}{|\mathcal{G}|})$; else $F_{4,j}^{(t)}(\mathbf{Z}^{1,0}) \le \widetilde{O}(\sigma_0)$. Combining them together, we complete the proof. $\square$

### D.3 Gradient Lemma

Starting with the gradient computation from Fact D.1:

$$-\nabla_{\mathbf{W}_{4,j,r,p}}\text{Loss}^1 = \frac{1}{2}\mathbb{E}\Big[\mathcal{E}_{4,j}\mathbf{sReLU}'(\Lambda_{4,j,r})\sum_{\mathbf{k}\in\mathcal{I}^{1,0}}\mathbf{Z}_{\mathbf{k},p}\Big],$$

we first consider the gradient for $j \in \tau(\mathcal{X})$

**Lemma D.1.** *For $j \in \tau(\mathcal{X})$, we have*

*(a) for $\mathbf{W}_{4,j,r,1}$, $s \in \tau(\mathcal{X})$*

 *(1) if $s = j$, $\langle -\nabla_{\mathbf{W}_{4,j,r,1}^{(t)}}\text{Loss}^1, e_s\rangle = \frac{1}{2}\mathbb{E}\Big[(1 - \mathbf{logit}_{4,j}^{(t)})\mathbf{sReLU}'(\Lambda_{4,j,r}^{(t)})\mathbb{1}_{\tau(x_1)=j}\Big]$;*

 *(2) $s \ne j$, $\langle -\nabla_{\mathbf{W}_{4,j,r,1}^{(t)}}\text{Loss}^1, e_s\rangle = \frac{1}{2}\mathbb{E}\Big[-\mathbf{logit}_{4,j}^{(t)}\mathbf{sReLU}'(\Lambda_{4,j,r}^{(t)})\mathbb{1}_{\tau(x_1)=s}\Big]$.*

*(b) for $\mathbf{W}_{4,j,r,2}$, $s = \tau(g)$ for $g \in \mathcal{G}$*

$$\langle -\nabla_{\mathbf{W}_{4,j,r,2}^{(t)}}\text{Loss}^1, e_s\rangle = \frac{1}{2}\mathbb{E}\Big[(1 - \mathbf{logit}_{4,j}^{(t)})\mathbf{sReLU}'(\Lambda_{4,j,r}^{(t)})\mathbb{1}_{\tau(x_1)=j,g_1=g}$$
$$- \mathbf{logit}_{4,j}^{(t)}\mathbf{sReLU}'(\Lambda_{4,j,r}^{(t)})\mathbb{1}_{\tau(x_1)\ne j,g_1=g}\Big].$$

*(c) for $\mathbf{W}_{4,j,r,p}$ with $p \in \{3, 4\}$, $s \in \tau(\mathcal{X})$*

 *(1) $s = j$, $\langle -\nabla_{\mathbf{W}_{4,j,r,3}}\text{Loss}^1, e_j\rangle = \frac{1}{2}\mathbb{E}\Big[-\mathbf{logit}_{4,j}^{(t)}\mathbf{sReLU}'(\Lambda_{4,j,r}^{(t)})\mathbb{1}_{\tau(x_0)=j}\Big]$;*

 *(2) $s \ne j$*

$$\langle -\nabla_{\mathbf{W}_{4,j,r,3}}\text{Loss}^1, e_s\rangle = \frac{1}{2}\mathbb{E}\Big[(1 - \mathbf{logit}_{4,j}^{(t)})\mathbf{sReLU}'(\Lambda_{4,j,r}^{(t)})\mathbb{1}_{\tau(x_0)=s,\tau(x_1)=j}$$
$$- \mathbf{logit}_{4,j}^{(t)}\mathbf{sReLU}'(\Lambda_{4,j,r}^{(t)})\mathbb{1}_{\tau(x_0)=s,j\notin\tau(\widehat{X})}\Big].$$

*(d) for $\mathbf{W}_{4,j,r,5}$, $s = \tau(y)$ for $g \in \mathcal{Y}$*

$$\langle -\nabla_{\mathbf{W}_{4,j,r,5}^{(t)}}\text{Loss}^1, e_s\rangle = \frac{1}{2}\mathbb{E}\Big[(1 - \mathbf{logit}_{4,j}^{(t)})\mathbf{sReLU}'(\Lambda_{4,j,r}^{(t)})\mathbb{1}_{\tau(x_1)=j,y_0=y}$$

$$- \mathbf{logit}_{4,j}^{(t)}\mathbf{sReLU}'\big(\Lambda_{4,j,r}^{(t)}\big)\mathbb{1}_{\tau(x_1)\neq j, y_0=y}\Big].$$

Move on to $j \notin \tau(\mathcal{X})$, we can obtain

**Lemma D.2.** *For $j \notin \tau(\mathcal{X})$, we have*

(a) *for $\mathbf{W}_{4,j,r,1}$, $s \in \tau(\mathcal{X})$,*

$$\langle -\nabla_{\mathbf{W}_{4,j,r,1}^{(t)}} \mathsf{Loss}^1, e_s \rangle = \frac{1}{2}\mathbb{E}\Big[ - \mathbf{logit}_{4,j}^{(t)}\mathbf{sReLU}'\big(\Lambda_{4,j,r}^{(t)}\big)\mathbb{1}_{\tau(x_1)=s}\Big].$$

(b) *for $\mathbf{W}_{4,j,r,2}$, $s = \tau(g)$ for $g \in \mathcal{G}$,*

$$\langle -\nabla_{\mathbf{W}_{4,j,r,2}^{(t)}} \mathsf{Loss}^1, e_s \rangle = \frac{1}{2}\mathbb{E}\Big[ - \mathbf{logit}_{4,j}^{(t)}\mathbf{sReLU}'\big(\Lambda_{4,j,r}^{(t)}\big)\mathbb{1}_{g_1=g}\Big].$$

(c) *for $\mathbf{W}_{4,j,r,p}$ with $p \in \{3,4\}$, $s \in \tau(\mathcal{X})$,*

$$\langle -\nabla_{\mathbf{W}_{4,j,r,p}^{(t)}} \mathsf{Loss}^1, e_j \rangle = \frac{1}{2}\mathbb{E}\Big[ - \mathbf{logit}_{4,j}^{(t)}\mathbf{sReLU}'\big(\Lambda_{4,j,r}^{(t)}\big)\mathbb{1}_{\tau(x_0)=s}\Big].$$

(d) *for $\mathbf{W}_{4,j,r,5}$, $s = \tau(y)$ for $g \in \mathcal{Y}$*

$$\langle -\nabla_{\mathbf{W}_{4,j,r,5}^{(t)}} \mathsf{Loss}^1, e_s \rangle = \frac{1}{2}\mathbb{E}\Big[ - \mathbf{logit}_{4,j}^{(t)}\mathbf{sReLU}'\big(\Lambda_{4,j,r}^{(t)}\big)\mathbb{1}_{y_0=y}\Big].$$

### D.4  Growth of Gamma

**Lemma D.3** (Growth). *Given $j \in \tau(\mathcal{X})$, suppose Induction D.1 holds at iteration t, when $\Phi_{4,j}^{(t)} \leq 0.01\log d$ or $\Gamma_{4,j}^{(t)} \leq \frac{0.01\log d}{m}$, then it satisfies*

$$\Gamma_{4,j}^{(t+1)} = \Gamma_{4,j}^{(t)} + \Theta\big(\frac{\eta}{d}\big)\mathbf{sReLU}'(\Gamma_{4,j}^{(t)}).$$

*Proof.* By Lemma D.1, we have

$$\langle -\nabla_{\mathbf{W}_{4,j,r,1}^{(t)}} \mathsf{Loss}^1, e_j \rangle = \frac{1}{2}\mathbb{E}\Big[(1 - \mathbf{logit}_{4,j}^{(t)})\mathbf{sReLU}'\big(\Lambda_{4,j,r}^{(t)}\big)\mathbb{1}_{\tau(x_1)=j}\Big].$$

By Claim D.2, when $\Phi_{4,j}^{(t)} \leq 0.01\log d$, $\mathbf{logit}_{4,j}^{(t)} = \frac{O(e^{0.01\log d})}{O(e^{0.01\log d})+d} \ll 1$ when $j = \tau(x_1)$; and combing with the fact that the event $\{\tau(x_1) = j\}$ happens with probability $\frac{1}{|\mathcal{X}|}$, we complete the proof. $\qquad\square$

Lemma D.3, combined with the growth of the tensor power method, immediately gives the following corollary.

**Lemma D.4.** *Suppose Induction D.1 holds for all iterations. Define threshold $\Lambda^- = \Theta(\frac{1}{m})$. Let $T_{1,j}$ be the first iteration so that $\Gamma_{4,j}^{(t)} \geq \Lambda^-$, and $T_1 \overset{def}{=} \Theta(\frac{d}{\eta\sigma_0^{q-2}})$. Then we have $T_1 \geq T_{1,j}$ for every $j \in \tau(\mathcal{X})$, i.e., for $t \geq T_1$, it satisfies $\Gamma_{4,j}^{(t)} \geq \Lambda^-$.*

**Lemma D.5** (Upper bound). *Suppose Induction D.1 holds for all iterations $< t$, we have $\Phi_{4,j}^{(t)} \leq \widetilde{O}(1)$, for $j \in \tau(\mathcal{X})$.*

*Proof.* We only need to consider the time $t \geq T_1$. Notice that the gradient descent update in Lemma D.1 gives

$$\langle -\nabla_{\mathbf{W}_{4,j,r,1}^{(t)}} \mathsf{Loss}^1, e_j \rangle = \frac{1}{2}\mathbb{E}\Big[(1 - \mathbf{logit}_{4,j}^{(t)})\mathbf{sReLU}'\big(\Lambda_{4,j,r}^{(t)}\big)\mathbb{1}_{\tau(x_1)=j}\Big]$$

Therefore, for sufficiently small $\eta$, we have

$$\Phi_{4,j}^{(t+1)} = \Phi_{4,j}^{(t)} + \sum_{r \in \mathcal{A}_{4,j}^{(t)}} \frac{\eta}{2}\mathbb{E}\Big[(1 - \mathbf{logit}_{4,j}^{(t)})\mathbf{sReLU}'\big(\Lambda_{4,j,r}^{(t)}\big)\mathbb{1}_{\tau(x_1)=j}\Big] + O(\varrho\log d)\cdot|\mathcal{A}_{4,j}^{(t+1)}\setminus\mathcal{A}_{4,j}^{(t)}|$$

$$= \Phi_{4,j}^{(t)} + \sum_{r \in \mathcal{A}_{4,j}^{(t)}} \frac{\eta}{2} \mathbb{E}\Big[\big(1 - \mathbf{logit}_{4,j}^{(t)}\big)\mathbf{sReLU}'\big(\Lambda_{4,j,r}^{(t)}\big)\mathbb{1}_{\tau(x_1)=j}\Big] + \frac{1}{\mathsf{polylog}d}.$$

When there exists $\widetilde{T}$, s.t., $\max_{j \in \tau(\mathcal{X})} \Phi_{4,j}^{(\widetilde{T})} > \Omega(\log^{1.5} d)$, by Induction D.1 and Claim D.1, given an input sequence $\mathbf{Z}^{1,0}$ with $\tau(x_1) = \widetilde{j} = \arg\max_{j \in \tau(\mathcal{X})} \Phi_{4,j}^{(\widetilde{T})}$, we have

$$F_{4,j}^{(\widetilde{T})}(\mathbf{Z}^{1,0}) \geq \sum_{r \in \mathcal{A}_{4,\widetilde{j}}^{(t)}} \Lambda_{4,\widetilde{j},r}^{(\widetilde{T})} \geq \left(1 - O\Big(\frac{1}{|\mathcal{G}|}\Big)\right)\Phi_{4,\widetilde{j}}^{(\widetilde{T})} - \widetilde{O}(\sigma_0) > \Omega(\log^{1.5} d).$$

Following the similar analysis as Claim D.2, $F_{4,j'}^{(\widetilde{T})}(\mathbf{Z}^{1,0}) \leq O(\frac{\Phi_{4,j'}^{(\widetilde{T})}}{|\mathcal{G}|})$ for other $j' \in \tau(\mathcal{X})$, and $F_{4,j'}^{(\widetilde{T})}(\mathbf{Z}) \leq o(1)$ for $j' \notin \tau(\mathcal{X})$, which implies $1 - \mathbf{logit}_{4,j}^{(\widetilde{T})} = e^{-\Omega(\log^{1.5} d)}$. Therefore, we derive that for $t \in [\widetilde{T} + 1, \frac{\mathsf{poly}d}{\eta})$,

$$\Phi_{4,j}^{(t)} \leq \Phi_{4,j}^{(\widetilde{T})} + \widetilde{O}(\mathsf{poly}d \cdot e^{-\Omega(\log^{1.5} d)}) + O(\rho \log d) \cdot m,$$

which implies $\Phi_{4,\widetilde{j}}^{(t)} \leq O(\log^{1.5} d)$ since $\varrho \ll \frac{1}{m \log d}$. $\qquad\square$

## D.5  Group and Value Correlations Are Not Large

**Lemma D.6.** *Suppose Induction D.1 holds for all iterations $< t$, then for any $j \in \tau(\mathcal{X})$ and $s = \tau(g), g \in \mathcal{G}$, we have*

$$\big|\langle \mathbf{W}_{4,j,r,2}^{(t)}, e_{\tau(g)}\rangle\big| \leq \widetilde{O}(\sigma_0) + O\Big(\frac{1}{|\mathcal{G}|}\Big) \max\Big\{ \langle \mathbf{W}_{4,j,r,1}^{(t)}, e_j\rangle, \min_{r' \in \mathcal{A}_{4,j^*}^{(t)}} \langle \mathbf{W}_{4,j^*,r',1}^{(t)}, e_{j^*}\rangle \Big\}$$

*Proof.* By Lemma D.1, we have

$$\langle -\nabla_{\mathbf{W}_{4,j,r,2}^{(t)}} \mathsf{Loss}^1, e_s\rangle$$
$$= \frac{1}{2}\mathbb{E}\Big[\big(1 - \mathbf{logit}_{4,j}^{(t)}\big)\mathbf{sReLU}'\big(\Lambda_{4,j,r}^{(t)}\big)\mathbb{1}_{\tau(x_1)=j,g_1=g} - \mathbf{logit}_{4,j}^{(t)}\mathbf{sReLU}'\big(\Lambda_{4,j,r}^{(t)}\big)\mathbb{1}_{\tau(x_1)\neq j,g_1=g}\Big].$$

Clearly, the positive gradient can be upper-bounded by $O\big(\frac{1}{|\mathcal{G}|}\langle -\nabla_{\mathbf{W}_{4,j,r,1}^{(t)}} \mathsf{Loss}^1, e_j\rangle\big)$. Moreover, for the negative gradient, by Claim D.1, we have a naive bound

$$\mathbf{sReLU}'\big(\Lambda_{4,j,r}^{(t)}\big)\big|_{\tau(x_1)\neq j,g_1=g} \leq O(1)\mathbf{sReLU}'\big(\Lambda_{4,j,r}^{(t)}\big)\big|_{\tau(x_1)=j,g_1=g}.$$

When $t \leq T_1$, by Claim D.2, we have $1 - \mathbf{logit}_{4,j}^{(t)}\big|_{j=\tau(x_1)} \geq \Omega(1)$ and $\mathbf{logit}_{4,j}^{(t)}\big|_{j\neq\tau(x_1)} \leq O(\frac{1}{d})$, which implies

$$\mathbb{E}\Big[\mathbf{logit}_{4,j}^{(t)}\mathbf{sReLU}'\big(\Lambda_{4,j,r}^{(t)}\big)\mathbb{1}_{\tau(x_1)\neq j,g_1=g}\Big] \leq O\Big(\frac{1}{|\mathcal{G}|}\Big)\langle -\nabla_{\mathbf{W}_{4,j,r,1}^{(t)}} \mathsf{Loss}^1, e_j\rangle.$$

Therefore, for $t \leq T_1$, we have

$$\big|\langle \mathbf{W}_{4,j,r,2}^{(t)}, e_{\tau(g)}\rangle\big| \leq \widetilde{O}(\sigma_0) + O\Big(\frac{1}{|\mathcal{G}|}\Big)\langle \mathbf{W}_{4,j,r,1}^{(t)}, e_j\rangle.$$

For $t \geq T_1$, notice that by Lemma D.4, $\mathcal{A}_{4,j'}^{(t)} \neq \emptyset$ for $j' \in \tau(\mathcal{X})$, thus for $r' \in \mathcal{A}_{4,j^*}^{(t)}$

$$\mathbf{sReLU}'\big(\Lambda_{4,j,r}^{(t)}\big)\big|_{\tau(x_1)\neq j,g_1=g} \leq \mathbf{sReLU}'\big(\Lambda_{4,j^*,r'}^{(t)}\big)\big|_{\tau(x_1)=j^*,g_1=g}.$$

Furthermore, $\mathbf{logit}_{4,j}^{(t)}\big|_{j\neq\tau(x_1)} \leq O(\frac{1}{d})(1 - \mathbf{logit}_{4,j^*}^{(t)}\big|_{j^*=\tau(x_1)})$, which implies

$$\mathbb{E}\Big[\mathbf{logit}_{4,j}^{(t)}\mathbf{sReLU}'\big(\Lambda_{4,j,r}^{(t)}\big)\mathbb{1}_{\tau(x_1)\neq j,g_1=g}\Big] \leq O\Big(\frac{1}{|\mathcal{G}|}\Big)\langle -\nabla_{\mathbf{W}_{4,j^*,r',1}} \mathsf{Loss}^1, e_{j^*}\rangle.$$

Due to the arbitrary of $r'$, we can conclude that

$$\left|\langle \mathbf{W}_{4,j,r,2}^{(t)}, e_{\tau(g)}\rangle\right| \leq \widetilde{O}(\sigma_0) + O\left(\frac{1}{|\mathcal{G}|}\right) \min_{r' \in \mathcal{A}_{4,j^*}^{(t)}} \langle \mathbf{W}_{4,j^*,r',1}^{(t)}, e_{j^*}\rangle.$$

$\square$

**Lemma D.7.** *Suppose Induction D.1 holds for all iterations $< t$, then for any $j \in \tau(\mathcal{X})$ and $s = \tau(y), y \in \mathcal{Y}$, we have*

$$\left|\langle \mathbf{W}_{4,j,r,5}^{(t)}, e_{\tau(y)}\rangle\right| \leq \widetilde{O}(\sigma_0) + O\left(\frac{1}{|\mathcal{Y}|}\right) \max\left\{\langle \mathbf{W}_{4,j,r,1}^{(t)}, e_j\rangle, \min_{r' \in \mathcal{A}_{4,j^*}^{(t)}} \langle \mathbf{W}_{4,j^*,r',1}^{(t)}, e_{j^*}\rangle\right\}.$$

*Proof.* The proof is similar to Lemma D.6, and we omit the details here. $\square$

### D.6 Off-diagonal Correlations Are Small

**Lemma D.8** (off-diagonal bound)**.** *Given $j \in \tau(\mathcal{X})$, suppose Induction D.1 holds at all iterations $< t$, for $s \in \tau(\mathcal{X}) \neq j$*

$$\left|\langle \mathbf{W}_{4,j,r,1}^{(t)}, e_s\rangle\right| \leq \widetilde{O}(\sigma_0).$$

*Proof.* By Lemma D.1

$$\langle -\nabla_{\mathbf{W}_{4,j,r,1}^{(t)}} \mathsf{Loss}^1, e_s\rangle = \frac{1}{2}\mathbb{E}\left[-\mathbf{logit}_{4,j}^{(t)} \mathbf{sReLU}'\big(\Lambda_{4,j,r}^{(t)}\big)\mathbb{1}_{\tau(x_1)=s}\right].$$

Notice that by Claim D.1,

$$\mathbf{sReLU}'\big(\Lambda_{4,j,r}^{(t)}\big)\big|_{\tau(x_1)=s} \leq O(1)\mathbf{sReLU}'\big(\Lambda_{4,s,r}^{(t)}\big)\big|_{\tau(x_1)=s},$$

combined with Claim D.2, $\mathbf{logit}_{4,j} \leq O(\frac{1}{d})(1 - \mathbf{logit}_{4,s}^{(t)})$ when $s = \tau(x_1)$, thus

$$\mathbb{E}\left[\mathbf{logit}_{4,j}^{(t)} \mathbf{sReLU}'\big(\Lambda_{4,j,r}^{(t)}\big)\mathbb{1}_{\tau(x_1)=s}\right] \leq \mathbb{E}\left[O(\frac{1}{d})(1 - \mathbf{logit}_{4,s}^{(t)})\mathbf{sReLU}'\big(\Lambda_{4,s,r}^{(t)}\big)\mathbb{1}_{\tau(x_1)=s}\right]$$

$$\leq O(\frac{1}{d})\langle -\nabla_{\mathbf{W}_{4,s,r,1}} \mathsf{Loss}, e_s\rangle.$$

From Induction D.1, we have

$$\left|\langle \mathbf{W}_{4,j,r,1}^{(t)}, e_s\rangle\right| \leq O(\frac{1}{d})\left|\langle \mathbf{W}_{4,s,r,1}^{(t)}, e_s\rangle\right| + \widetilde{O}(\sigma_0) \leq \widetilde{O}(\frac{1}{d}) + \widetilde{O}(\sigma_0) = \widetilde{O}(\sigma_0).$$

$\square$

**Lemma D.9.** *Given $j \in \tau(\mathcal{X})$, suppose Induction D.1 holds at all iterations $< t$, we have*

$$\left|\langle \mathbf{W}_{4,j,r,p}^{(t)} \mathsf{Loss}^1, e_s\rangle\right| \leq \widetilde{O}(\sigma_0), \quad \text{for } p \in \{3,4\} \text{ and all } s \in \tau(\mathcal{X})$$

*Proof.* When $s = j$, we have

$$\langle -\nabla_{\mathbf{W}_{4,j,r,p}} \mathsf{Loss}^1, e_j\rangle = \frac{1}{2}\mathbb{E}\left[-\mathbf{logit}_{4,j}^{(t)} \mathbf{sReLU}'\big(\Lambda_{4,j,r}^{(t)}\big)\mathbb{1}_{\tau(x_0)=j}\right]$$

$$= \frac{1}{2}\mathbb{E}\left[-\mathbf{logit}_{4,j}^{(t)} \mathbf{sReLU}'\big(\Lambda_{4,j,r}^{(t)}\big)\sum_{s \neq j}\mathbb{1}_{\tau(x_0)=j, \tau(x_1)=s}\right].$$

Therefore, we can bound the above gradient in the similar way as the off-diagonal case, and obtain

$$\left|\langle \mathbf{W}_{4,j,r,p}^{(t)}, e_j\rangle\right| \leq O(\frac{1}{d})\max_{s \in \tau(\mathcal{X})}\left|\langle \mathbf{W}_{4,s,r,1}^{(t)}, e_s\rangle\right| + \widetilde{O}(\sigma_0) \leq \widetilde{O}(\sigma_0).$$

When $s \neq j$,

$$\langle -\nabla_{\mathbf{W}_{4,j,r,p}} \mathsf{Loss}^1, e_s\rangle$$

$$= \frac{1}{2}\mathbb{E}\Big[(1 - \mathbf{logit}_{4,j}^{(t)})\mathbf{sReLU}'\big(\Lambda_{4,j,r}^{(t)}\big)\mathbb{1}_{\tau(x_0)=s,\tau(x_1)=j} - \mathbf{logit}_{4,j}^{(t)}\mathbf{sReLU}'\big(\Lambda_{4,j,r}^{(t)}\big)\mathbb{1}_{\tau(x_0)=s,j\notin\tau(\widehat{X})}\Big].$$

Noticing that $\{\tau(x_0) = s, \tau(x_1) = j\}$ happens with probability $\frac{1}{|\mathcal{X}|(|\mathcal{X}|-1)}$, thus the positive gradient can be upper bounded by $O(\frac{1}{d}) \cdot |\langle -\nabla_{\mathbf{W}_{4,j,r,1}}\mathsf{Loss}^1, e_j\rangle|$. Furthermore, the negative part can be upper bounded in the similar way as previous off-diagonal negative gradient. Putting it together, we complete the proof. $\qquad\square$

### D.7 Non-target Correlations Are Negligible

**Lemma D.10.** *Suppose Induction D.1 holds at all iterations $< t$, for $j' \notin \tau(\mathcal{X})$, for $p \in [5]$ and $s \in [d]$*

$$\big|\langle \mathbf{W}_{4,j',r,p}^{(t)}, e_s\rangle\big| \leq \widetilde{O}(\sigma_0).$$

*Proof.* By Lemma D.2, $\langle \mathbf{W}_{4,j',r,p}^{(t)}, e_s\rangle$ for $p \in \{1, 3, 4\}$ and $s \in \tau(\mathcal{X})$ can be bounded in the similar way previous off-diagonal negative gradient.

We can observe that for $j' \notin \tau(\mathcal{X})$, all the non-zero gradient on the different directions are negative gradient, which implies $\langle \mathbf{W}_{4,j',r,p}^{(t)}, e_s\rangle \leq \langle \mathbf{W}_{4,j',r,p}^{(0)}, e_s\rangle = \widetilde{O}(\sigma_0)$. Moreover, $\Lambda_{4,j',r}^{(t)} \leq \widetilde{O}(\sigma_0)$ is also non-increasing.

For $s = \tau(g), g \in \mathcal{G}$, whenever $\langle \mathbf{W}_{4,j',r,2}^{(t)}, e_s\rangle$ reaches $-3\mu$, we have $\Lambda_{4,j',r}^{(t)}\big|_{g_1=g} \leq -3\mu + \frac{5}{2}\mu + \widetilde{O}(\sigma_0) \leq 0$, and thus $\langle -\nabla_{\mathbf{W}_{4,j,r,2}^{(t)}}\mathsf{Loss}^1, e_s\rangle = \frac{1}{2}\mathbb{E}\Big[-\mathbf{logit}_{4,j}^{(t)}\mathbf{sReLU}'\big(\Lambda_{4,j,r}^{(t)}\big)\mathbb{1}_{g_1=g}\Big] = 0$, which implies $\langle \mathbf{W}_{4,j',r,2}^{(t)}, e_s\rangle \geq -3\mu$. Hence, $|\langle \mathbf{W}_{4,j',r,2}^{(t)}, e_s\rangle| \leq \widetilde{O}(\sigma_0)$. Following the similar argument, we can prove the result for $\langle \mathbf{W}_{4,j',r,5}^{(t)}, e_s\rangle$ for $s \in \tau(\mathcal{Y})$. $\qquad\square$

### D.8 Convergence

**Lemma D.11.** *For $|\mathcal{G}| \geq |\mathcal{Y}| \geq \Omega(\frac{\log\log d}{\log\log\log d})$, $\mathsf{polylog}d \geq m \geq |\mathcal{Y}|$, $\varrho \ll \frac{1}{m\log d}$ and sufficiently small $\eta \leq \frac{1}{\mathsf{poly}d}$, Induction D.1 holds for all iterations $t \leq T = \frac{\mathsf{poly}d}{\eta}$.*

*Proof.* Putting the results in Lemmas D.5 to D.8 and D.10, we can directly establish the results in Induction D.1. $\qquad\square$

**Lemma D.12** (Convergence). *For sufficiently large $T_1 \leq t = \frac{\mathsf{poly}d}{\eta}$, we have*

(a) *Objective convergence:* $\mathsf{Loss}^1 \leq \frac{1}{\mathsf{poly}d}$;

(b) *Successful learning of diagonal feature:* $\Phi_{4,j}^{(t)} \geq \Omega(\log d)$ *for any $j \in \tau(\mathcal{X})$.*

*Proof.* Assuming for some sufficiently large constant $n > 0$, $\mathbb{E}[(1 - \mathbf{logit}_{4,j^*}^{(t)}) \mid \tau(x_1) = j^*] \geq \Omega(\frac{1}{d^n})$ for $t \in (T_1, T_1 + \frac{d^{n+1}\log^2 d}{\eta}]$ then by Lemma D.1, we have

$$\Gamma_{4,j^{(*)}}^{(T_1 + \frac{d^2\log^2 d}{\eta})} \geq \Omega\Big(\frac{\eta}{d^{n+1}}\Big) \cdot \frac{d^{n+1}\log^2 d}{\eta} + \Gamma_{4,j^{(*)}}^{(t)} \geq \Omega(\log^2 d),$$

which contradicts with $\Gamma_{4,j^{(*)}}^{(t)} \leq \Phi_{4,j^{(*)}}^{(t)} \leq O(\log^{1.5} d) = \widetilde{O}(1)$ in the polynomial time. This implies after sufficiently large iteration $t$, we must have $\mathbb{E}[(1 - \mathbf{logit}_{4,j}^{(t)}) \mid \tau(x_1) = j] \leq O(\frac{1}{d^n})$ for $j \in \tau(\mathcal{X})$. Hence

$$\begin{aligned}
\mathsf{Loss}^1 = \mathbb{E}[-\log\mathbf{logit}_{4,\tau(x_1)}^{(t)}] &= \sum_{j\in\tau(\mathcal{X})}\mathbb{E}[-\log\mathbf{logit}_{4,j}^{(t)}\mathbb{1}_{\tau(x_1)=j}] \\
&\leq \sum_{j\in\tau(\mathcal{X})}\mathbb{E}[O(1)(1 - \mathbf{logit}_{4,j}^{(t)})\mathbb{1}_{\tau(x_1)=j}] \\
&\hphantom{=====================} (\mathbf{logit}_{4,j}^{(t)} \text{ is very close to 1}) \\
&\leq O\Big(\frac{1}{\mathsf{poly}d}\Big).
\end{aligned}$$

By Claim D.2, at the time of convergence, we must have $\Phi_{4,j}^{(t)} \geq \Omega(\log d)$. $\qquad\square$

# E  Learning Simply Transitive Actions

## E.1  Preliminaries

**Definition E.1** (Combinations $\phi, \Phi, \Phi^\dagger$). Let $\mathcal{G}, \mathcal{Y}$ be defined as in Definition 2.2. For any pair $(g, y) \in \mathcal{G} \times \mathcal{Y}$, we call $\phi = (g, y)$ a **combination**. We write $\phi_1 = g$ and $\phi_2 = y$ to denote the corresponding components. We further define the set of **correct combinations** $\Phi^\star$ by

$$\Phi_j^\star = \{\phi = (g, y) \mid j = \tau(\alpha(g, y))\}, \ \forall j \in \tau(\mathcal{Y}); \qquad \Phi = \bigcup_{j \in \tau(\mathcal{Y})} \Phi_j^\star \qquad (24)$$

The combinations $\phi \in \Phi_j^\star$ are the ones that can be composed to get $\widetilde{y} = \tau^{-1}(j) \in \mathcal{Y}$. We also define the set of **incorrect combinations** $\Phi_j^\dagger$ for each $j \in \tau(\mathcal{Y})$ to be

$$\Phi_j^\dagger = \{\phi \in \Phi \setminus \Phi_j^\star\} \qquad (25)$$

For simply transitive action $\alpha : \mathcal{G} \times \mathcal{Y} \to \mathcal{Y}$, any combination $\phi \in \Phi_j^\dagger$ is a *incorrect* pair of $g$ and $y$. That is, it holds that any $y' \neq y \in \mathcal{Y}$ and $g' \neq g \in \mathcal{G}$ satisfy $\alpha(g, y') = \alpha(y, g') \neq \tau^{-1}(j)$.

For each combination, we define the following neural-features:

**Definition E.2** (combination features $\psi, \Psi, \Psi^\star, \Psi^\dagger$). Given a combination $\phi = (g, y) \in \Phi$, token index $j \in \tau(\mathcal{Y})$ and neuron index $r \in [m]$, we define

$$V_{j,r}(g) = \langle \mathbf{W}_{5,j,r,2}, e_g \rangle, \quad V_{j,r}(y) = \langle \mathbf{W}_{5,j,r,5}, e_y \rangle, \quad V_{j,r}(\phi) := \frac{1}{2}(V_{j,r}(g) + V_{j,r}(y))$$

and we call $V_{j,r}(\phi)$ the *features for combination* $\phi$ in neuron $(j, r) \in \tau(\mathcal{Y}) \times [m]$. We write $\psi = (j, r, \phi)$ and $V_\psi \equiv V_{j,r}(\phi)$ to make the notation concise. Similar to above, we further define

$$\Psi := \{\psi = (j, r, \phi) \times \tau(\mathcal{Y}) \times [m] \times \Phi\}$$

and $\Psi^\star$ that contains the desirable features for each class $j \in \tau(\mathcal{Y})$:

$$\Psi_{j,r}^\star = \{\psi = (j, r, \phi) \mid \phi \in \Phi_j^\star\}, \quad \Psi^\star = \bigcup_{(j,r) \in \tau(\mathcal{Y}) \times [m]} \Psi_{j,r} \qquad (26)$$

and $\Psi^\dagger$ that contains the incorrect feature combinations:

$$\Psi_{j,r}^\dagger = \{\psi = (j, r, \phi) \mid \phi \in \Phi \setminus \Phi_j^\star\}, \quad \Psi^\dagger = \bigcup_{(j,r) \in \tau(\mathcal{Y}) \times [m]} \widehat{\Psi}_{j,r} \qquad (27)$$

**Events of combination appearance.** Let us first define some useful notations. For a combination $\phi = (g, y)$, we write $\mathcal{H}_\phi$ to denote the event when $\phi$ appears in the sequence $\mathbf{Z}^1$:

$$\mathcal{H}_\phi := \{(g_1, y_0) = \phi\}, \qquad \mathcal{H}_{\phi_1} = \mathcal{H}_g = \{g_1 = g\}, \qquad \mathcal{H}_{\phi_2} = \mathcal{H}_y = \{y_0 = y\}.$$

Further, we write

$$\mathcal{H}_{\phi,1}^\dagger = \{g_1 \neq g, y_0 = y\}, \quad \mathcal{H}_{\phi,2}^\dagger = \{g_1 = g, y_0 \neq y\}, \quad \mathcal{H}_\phi^\dagger = \mathcal{H}_{\phi,1}^\dagger \cup \mathcal{H}_{\phi,2}^\dagger$$

to denote the event where $\phi$ did not appear but its group element or value is the same.

Finally we define a notion called *learning curriculum*, that sits at the center of our proof.

**Definition E.3** (Learning curriculum). We define an order on the set $\Sigma$, defined by the following process: Let $\Sigma_0 = \Psi$. At each $i \in [n_y^2]$, we choose

$$\psi_i = \arg\max_{\psi \in \Sigma_i} V_\psi^{(0)}$$

Let's write $\psi_i = (j, r, \phi)$ where $\phi = (g, y)$, then we define $\Sigma_{i+1}$ by excluding the following features:

1. Exclude the confusing combinations in neuron $(j, r)$;

$$\Sigma_{\psi_i}^{\dagger,1} \equiv \Sigma_i^{\dagger,1} = \{\psi' = (j, r, \phi') \in \Sigma_i, \phi' = (g', y') \mid (g' = g) \ \mathbf{XOR} \ (y' = y)\}$$

2. Exclude the unselected combinations in neuron $(j, r)$;

$$\Sigma^{\dagger,2}_{\psi_i} \equiv \Sigma^{\dagger,2}_i = \{\psi' = (j, r, \phi') \in \Sigma_i, \phi' = (g', y') \mid g' \neq g, y' \neq y\}$$

3. Exclude the feature indices of the same combination in other neurons $(j, r'), r' \neq r$;

$$\Sigma^{\dagger,3}_{\psi_i} \equiv \Sigma^{\dagger,3}_i = \{\psi' = (j, r', \phi) \in \Sigma_i, \forall r' \neq r \in [m]\}.$$

Which returns $\Sigma_{i+1} = \Sigma_i \setminus \{\psi = (j, r', \phi') \in \Sigma_i \mid \text{either } r' = r \text{ or } \phi' = \phi\}$. Iterate over the whole set $\Psi$, we will arive at $\Sigma_{n_y^2} = \varnothing$. This gives rise to an ordered sequence $\Sigma^\star$ and a set $\Sigma^\dagger$:

$$\Sigma^\star := (\psi_1, \psi_2, \dots, \psi_{n_y^2}); \qquad \Sigma^\dagger := \bigcup_{\psi \in \Sigma^\star} \Sigma^\dagger_\psi, \qquad \Sigma^\dagger_\psi := \Sigma^{\dagger,1}_\psi \cup \Sigma^{\dagger,2}_\psi \cup \Sigma^{\dagger,3}_\psi$$

By our construction $V^{(0)}_{\psi_i} \geq V^{(0)}_{\psi_{i+1}}$. We write $\psi \prec_\Sigma \psi'$ to denote that $\psi$ is ahead of $\psi'$ in $\Sigma^\star$.

Intuitively, $\Sigma^\star$ encodes the order at which the features $V_\psi$ grow in magnitude and $\Sigma^\dagger$ contains all the unlearned features during the process.

Throughout the analysis of the FFN layer, we further make the following assumptions.

**Assumption E.1.** For $(j, r, p) \in [d] \times [m] \times [5]$, we fix $\mathbf{W}^{(t)}_{5,j,r,p} \equiv \mathbf{W}^{(0)}_{5,j,r,p}$ for $p \in \{1, 3, 4\}$ at initialization for simplicity of proof.

**Assumption E.2.** We use a modified smoothed ReLU as our activation function. This technical assumption is to avoid many pathologies appearing in the learning dynamics.

$$\mathbf{sReLU}(x) := \begin{cases} \varpi B & x \leq -B \\ -\varpi x & x \in (-B, -\varpi] \\ \frac{1}{2}x^2 & x \in (-\varpi, 0]; \\ x^q/(\varrho^{q-1}q) & x \in (0, \varrho]; \\ x - \varrho\left(1 - \frac{1}{q}\right) & x \in (\varrho, B] \\ B - \varrho\left(1 - \frac{1}{q}\right) & x > B \end{cases}$$

where $q = O(1)$ is a large even integer and $\varrho = \Theta(1/\mathrm{polylog}(d)), \varpi \in (d^2\mu^{q-1}, \frac{\lambda}{d^{q/3}}), B = C\log d, C = \Theta(1) \in (5, \frac{q-1}{3})$ and $\lambda = \frac{d-1}{d-1+e^B}$.

### E.1.1 Theorem Statement

We will try to prove the following theorem:

**Theorem E.1** (Learning Simply Transitive Actions). *Suppose $F^{(t)}$ is returned by Algorithm 1 at $t = T_1$, and let $\delta_1 = d^{c_1}\mu, \delta_2 = \varpi^{\frac{1}{q-1}}$, it holds that with probability $\geq 1 - o(1)$:*

A. *For all $\psi \in \Sigma^\star$, we have $V^{(T_1)}_\psi = B \pm O(\delta_1)$;*

B. *For any $\psi \in \Sigma^{\dagger,1}$, $|V^{(T_1)}_\psi| \leq \widetilde{O}(\delta_2)$;*

C. *For any $\psi \in \Sigma^{\dagger,2}$, $V^{(T_1)}_\psi \leq -B + O(\delta_1)$;*

D. *For any $\psi \in \Sigma^{\dagger,3}$, $V^{(T_1)}_\psi \leq O(\delta_1)$;*

E. *For $\psi = (j, r, (g, y)) \in \Sigma^\star$, $|V^{(t)}_{j,r}(g) - V^{(t)}_{j,r}(y)| \leq \widetilde{O}(\delta_2)$.*

In fact, we can decompose the statement of theorem E.1 into the following statement for every feature $\psi \in \Sigma^\star$.

**Definition E.4** (Feature Shape). Let $\delta = (\delta_1, \delta_2)$ be a tuple of error parameters. Let $\psi \in \Sigma^\star$ be a feature in the learning curriculum. We say the feature $\psi$ reached *feature shape* $\mathcal{F}_\psi(\delta)$ with error $\delta$ if:

1. $\mathcal{F}_{\psi,1}(\delta_1)$: $V^{(t)}_\psi \geq B - O(\delta_1)$;

2. $\mathcal{F}_{\psi,2}(\delta_2)$: For any $\psi' \in \Sigma^{\dagger,1}_\psi$, it holds that $|V^{(t)}_{\psi'}| \leq \widetilde{O}(\delta_2)$;

3. $\mathcal{F}_{\psi,3}(\delta_1)$: For any $\psi' \in \Sigma^{\dagger,2}_\psi$, it holds that $V^{(t)}_{\psi'} \leq -B \pm O(\delta_1)$;

4. $\mathcal{F}_{\psi,4}(\delta_1)$: For any feature $\psi' \in \Sigma_\psi^{\dagger,3}$, it holds that $V_{\psi'}^{(t)} \leq O(\delta_1)$.

5. $\mathcal{F}_{\psi,5}(\delta_2)$: Writing $\psi = (j, r, (g, y))$, then it holds that $|V_{j,r}^{(t)}(g) - V_{j,r}^{(t)}(y)| \leq \widetilde{O}(\delta_2)$;

Obviously if we proved the condition $\mathcal{F}_\psi(\delta)$ of all $\psi \in \Sigma^\star$ are reached for $\delta = (d^{c_1}\mu, \varpi^{\frac{1}{q-1}})$ at $T_1$, then theorem E.1 is proven. We do this by following a induction process sequentially for each $\psi \in \Sigma^\star$, and provide a guarantee that all features in $\Sigma^\star$ reach the claimed convergence condition together at $t = T_1$.

### E.1.2 Induction Hypotheses and Phase Decomposition

To prove theorem E.1, or equivalently, to prove the convergence condition in Definition E.4 is reached for every feature $\psi \in \Sigma^\star$, we shall charaterize the dynamics of each feature and prove that they eventually arrive at the desired shape.

**Phase Decomposition.** Let us define the following timestamps for different phases of learning each $\psi = (j, r, \phi) \in \Psi \cup \widehat{\Psi}$. Let $c_1 = \frac{1}{1000}$ be a small constant:

(a) Phase I: $t \in [0, T_\psi^1]$, where $T_\psi^1 := \min\{t \geq 0 \mid V_\psi^{(t)} \geq d^{c_1}\mu\}$;

(b) Phase II.1: $t \in (T_\psi^1, T_\psi^{2,1}]$, where $T_\psi^{2,1} := \min\{t \geq 0 \mid V_\psi^{(t)} \geq \frac{1}{2}\log d\}$;

(c) Phase II.2: $t \in (T_\psi^{2,1}, T_\psi^{2,2}]$, where $T_\psi^{2,2} := \min\{t \geq 0 \mid \mathbb{E}[\textbf{logit}_{5,j}^{(t)}\mathbb{1}_{\mathcal{H}_\phi}] \geq 1 - \frac{1}{\sqrt{d}}\}$;

(d) Phase II.3: $t \in (T_\psi^{2,2}, T_\psi^2]$, where $T_\psi^2 = \max\{T_\psi^{2,3}, T_\psi^{2,2} + \Omega(\frac{1}{\eta(d^{c_1}\mu)^{q-2}})\}$ and $T_\psi^{2,3}$ is defined as:

$$T_\psi^{2,3} := \min\left\{t \geq T_\psi^{2,2} \mid \mathcal{F}_\psi(\delta_1, \delta_2) \text{ holds, where } \delta_1 = d^{c_1}\sigma_0, \delta_2 = (\frac{n_y^2 d\varpi}{\lambda})^{\frac{1}{q-1}}\right\}$$

(e) Phase III.1: $t \in (T_\psi^2, T_{1,1}]$, where $T_{1,1} = T_{\psi_{n_y^2}}^2$ and $\psi_{n_y^2}$ is the last feature in $\Sigma^\star$. This is the convergence phase where the rest of the features are learned and the feature converge to a perfect shape.

(f) Phase III.2: $t \in (T_{1,1}, T_1]$, the end phase where all features are in perfect shape and have stablized.

Now we state the following induction hypotheses which naturally results in theorem E.1.

**Induction E.1** (Simply Transitive Actions, All Phases). *For all $t \leq T_1$, the following holds:*

(a) $V_\psi^{(t)} + b_{i,j,r} \geq \Omega(\mu)$.

(b) $T_{\psi'}^2 < T_\psi^1$ if $\psi' \prec_\Sigma \psi$ for $\psi, \psi' \in \Sigma^\star$, i.e., the intervals $\{[T_\psi^1, T_\psi^2]\}_{\psi \in \Sigma^\star}$ are non-overlapping;

In order to prove Induction E.1 and theorem E.1, we introduce the following feature based induction hypothesis to prove that $\mathcal{F}_\psi$ defined in Definition E.4 holds for some error parameters at $t = T_\psi^2$.

**Induction E.2** (Induction for Individual Feature). *Let $\psi = (j, r, \phi) \in \Sigma^\star, \phi = (g, y)$, at $t \leq T_1$, the following holds:*

(a) *At $t \leq T_\psi^1$, any $g \neq g', y' \neq y$ has $V_{j,r}(g'), V_{j,r}(y') \leq \widetilde{O}(\mu)$.*

(b) *During $t \leq T_\psi^1$, there are at most $\widetilde{O}(\sqrt{d}/\eta)$ iterations where $\textbf{logit}^{(t)} \geq \frac{1}{\sqrt{d}}$ conditioned on $\mathcal{H}_\phi$.*

(c) *Let $\psi' \prec \psi \in \Sigma^\star$, the feature shape $\mathcal{F}_{\psi'}(d^{c_1}\sigma_0, (\frac{n_y^2 d\varpi}{\lambda})^{\frac{1}{q-1}})$ holds throughout $t \in [T_\psi^1, T_1]$;*

(d) *Let $\psi' \succ \psi \in \Sigma^\star$, then $V_\psi^{(t)} \leq \widetilde{O}(\sigma_0)$ throughout $t \in [0, T_\psi^2]$.*

### E.1.3 Technical Calculations

Let us recall some facts about gradient computation for $\mathbf{W}$.

**Fact E.1** (gradient computation). The gradient with respect to $\mathbf{W}_{i,j,r,p}$ for $i = 5$, $j \in [d]$, $r \in [m]$, $p \in [5]$ and $v \in \mathcal{V}$ is

$$\langle -\nabla_{\mathbf{W}_{5,j,r,p}}\text{Loss}, e_v\rangle = \frac{1}{2}\mathbb{E}\Big[\mathcal{E}_{5,j}\mathbf{sReLU}'(\Lambda_{5,j,r})\sum_{\mathbf{k}\in\mathcal{I}^{1,0}}\langle\mathbf{Z}_{\mathbf{k},p}, e_v\rangle\Big] \tag{28}$$

where $\mathcal{E}_{5,j} = \mathbb{1}_{v=Z_{\mathsf{ans},1,p}} - \mathbf{logit}_{5,j}$ is the loss derivatives. We compute more precise expressions when $j$, $p$ and $v$ varies:

(a) when $p = 2$, for $g \in \mathcal{G}$, let $\phi = (g, y) \in \Phi_j^\star$ be the combination with $\phi_1 = g$, we have for all $j \in \tau(\mathcal{Y}), r \in [m]$,

$$\langle -\nabla_{\mathbf{W}_{5,j,r,2}}\text{Loss}, e_g\rangle = \frac{1}{2}\mathbb{E}\Big[(1 - \mathbf{logit}_{5,j})\mathbf{sReLU}'(\Lambda_{5,j,r})\mathbb{1}_{\mathcal{H}_\phi}\Big]$$
$$- \frac{1}{2}\mathbb{E}\Big[\mathbf{logit}_{5,j}\mathbf{sReLU}'(\Lambda_{5,j,r})\mathbb{1}_{\mathcal{H}_{\phi,2}^\dagger}\Big] \tag{29}$$

(b) when $p = 5$, for $y \in \mathcal{Y}$, let $\phi = (g, y) \in \Phi_j^\star$ be the combination with $\phi_2 = y$, we have for all $j \in \tau(\mathcal{Y}), r \in [m]$,

$$\langle -\nabla_{\mathbf{W}_{5,j,r,5}}\text{Loss}, e_y\rangle = \frac{1}{2}\mathbb{E}\Big[(1 - \mathbf{logit}_{5,j})\mathbf{sReLU}'(\Lambda_{5,j,r})\mathbb{1}_{\mathcal{H}_\phi}\Big]$$
$$- \frac{1}{2}\mathbb{E}\Big[\mathbf{logit}_{5,j}\mathbf{sReLU}'(\Lambda_{5,j,r})\mathbb{1}_{\mathcal{H}_{\phi,1}^\dagger}\Big] \tag{30}$$

(c) For any combination of $p$, $v$ not included above, we have due to our assumptions of the distribution and update rule:

$$\langle -\nabla_{\mathbf{W}_{5,j,r,p}}\text{Loss}, e_v\rangle \equiv 0$$

**Definition E.5** ($\Gamma$-notations). To simplify gradient expression, we define the following notation: let $\psi = (j, r, \phi) \in \Psi$, we write

$$\Gamma_{j,r,\phi}^{+,(t)} = \Gamma_\psi^{+,(t)} := \mathbb{E}\Big[(1 - \mathbf{logit}_{5,j})\mathbf{sReLU}'(\Lambda_{5,j,r})\mathbb{1}_{\mathcal{H}_\phi}\Big]$$
$$\Gamma_{j,r,\phi}^{-,(t)} = \Gamma_\psi^{-,(t)} := \mathbb{E}\Big[\mathbf{logit}_{5,j}\mathbf{sReLU}'(\Lambda_{5,j,r})\mathbb{1}_{\mathcal{H}_\phi^\dagger}\Big] \tag{31}$$

we can further define $\Gamma$ for each $g \in \mathcal{G}$ and $y \in \mathcal{Y}$: Let $v \in \mathcal{G} \cup \mathcal{Y}$, we define

$$\Gamma_{j,r,v}^{+,(t)} = \Gamma_{\psi,v}^{+,(t)} := \mathbb{E}\Big[(1 - \mathbf{logit}_{5,j})\mathbf{sReLU}'(\Lambda_{5,j,r})\mathbb{1}_{\mathcal{H}_v}\Big]$$
$$\Gamma_{j,r,v}^{-,(t)} = \Gamma_{\psi,v}^{-,(t)} := \mathbb{E}\Big[\mathbf{logit}_{5,j}\mathbf{sReLU}'(\Lambda_{5,j,r})\mathbb{1}_{\mathcal{H}_v^\dagger}\Big] \tag{32}$$

This allows us to define the following gradient condition:

**Definition E.6** (gradient condition). At $t \leq T_1$, letting $\psi = (j, r, \phi) \in \Psi$, we write $\mathcal{C}_\psi(\delta)$ for $\delta > 0$ to denote that the gradient update for $V_\psi$ is smaller than $\delta$, formally:

$$\mathcal{C}_{j,r,v}(\delta) \text{ holds} \implies |\Gamma_{j,r,v}^{+,(t)} - \Gamma_{j,r,v}^{-,(t)}| \leq \delta$$
$$\mathcal{C}_{j,r,v}^+(\delta) \text{ holds} \implies |\Gamma_{j,r,v}^{+,(t)}| \leq \delta$$
$$\mathcal{C}_{j,r,v}^-(\delta) \text{ holds} \implies |\Gamma_{j,r,v}^{-,(t)}| \leq \delta$$

We further write

$$\mathcal{C}_\psi(\delta) \text{ holds} \implies \mathcal{C}_{j,r,g}(\delta) \wedge \mathcal{C}_{j,r,y}(\delta) \text{ holds}$$
$$\mathcal{C}_\psi^+(\delta) \text{ holds} \implies \mathcal{C}_{j,r,g}^+(\delta) \wedge \mathcal{C}_{j,r,y}^+(\delta) \text{ holds}$$
$$\mathcal{C}_\psi^-(\delta) \text{ holds} \implies \mathcal{C}_{j,r,g}^-(\delta) \wedge \mathcal{C}_{j,r,y}^-(\delta) \text{ holds}$$

Now we proceed to characterize the lower bound of logits

**Fact E.2** (logit lower bound). We list some basic facts about the architecture Definition 3.5 here.

1. For any $Z^1 \in \text{supp}(\mathcal{D}^1)$, we have

$$\lambda := \min_{i,j,F,Z}(1 - \textbf{logit}_{i,j}(F, Z)) = \frac{d-1}{d-1+e^B}$$

2. Suppose for some $i \in [5]$, $F_{i,j} = o(1)$ for all $j \notin \tau(\mathcal{Y})$, then

$$\min_{i,j,F,Z} \textbf{logit}_{i,j}(F, Z) = \frac{1}{(1+o(1))d + n_y e^B} = O(\frac{\lambda}{n_y d})$$

## E.2 Phase I: Feature Emergence and Competition

We shall prove the following properties at initialization.

**Fact E.3** (initialization range). At $t = 0$, the following holds with probability $\geq 1 - o(1)$:

(a) $|\langle \mathbf{W}_{5,j,r,p}^{(0)}, e_v \rangle| \leq O(\sigma_0 \sqrt{\log d})$ For all $j, r, p$ and $v \in \mathcal{V}$;

(b) $V_\psi^{(0)} \geq V_{\psi'}^{(0)} + \gamma$, where $\gamma = \Omega(\frac{\sigma_0}{n_y^4 m^2 \log d})$, for any pair $\psi \prec \psi' \in \Sigma^\star$ in Def. E.3;

*Proof.* Fact E.3a can easily verified from our initialization $\mathbf{W}_{i,j,r,p}^{(0)} \sim \mathcal{N}(0, \sigma_0 I_d)$. For Fact E.3b, we give a straightforward proof that every pair of $V_\psi^{(0)}, \psi \in \Sigma^\star$ has a gap of $\frac{\sigma_0}{n_y^4 m^2 \log d}$. First note that $V_\psi^{(0)}$ of different $\psi \in \Sigma^\star$ are independent and identically distributed on the randomness of $\mathbf{W}^{(0)}$, due to the orthogonality of embeddings $e_v, v \in \mathcal{V}$. Then, by the basic property of a Gaussian variable (notice that $V_\psi^{(0)} - V_{\psi'}^{(0)}$ is also Gaussian with variance $2\sigma_0$) (all though different pairs could be dependent), we have with probability $1 - \frac{1}{n_y^4 m^2 \log d}$ that their gap is at least $\gtrsim \frac{\sigma_0}{n_y^4 m^2 \log d}$ for each pair. Then by a union bound over $O(m^2 n_y^4)$-many all possible pairs we can conclude the proof. $\square$

We give another characterization of the initial activations.

**Fact E.4** (activation magnitude). At $t = 0$, with high probability it holds that

$$\Lambda_{5,j,r}^{(0)} = (1 + O(\frac{1}{\sqrt{\log d}}))\mu = \Theta(\sigma \log d) \ll \varrho$$

and thus $\textbf{sReLU}(\Lambda_{5,j,r}) = \Theta(\frac{1}{q}\mu^q)$, $\textbf{sReLU}'(\Lambda_{5,j,r}) = \Theta(\mu^{q-1})$ and $F_{5,j} = \Theta(\frac{m}{q}\mu^q)$.

*Proof.* Combining Definitions 3.4 and 3.5, Assumption 3.3, and Fact E.3 gives the fact. $\square$

We establish some properties in Phase I.

**Lemma E.1** (Key properties in Phase I). *Let $\psi = (j, r, (g, y)) \in \Sigma^\star$, and assume Inductions E.1 and E.2 holds at $t \leq T_\psi^1$, then*

(a) $|V_{j,r}^{(t)}(v)| \leq O(\mu/\sqrt{\log d})$ *if $v \neq g, v \neq y$;*

(b) $\Lambda_{5,j,r}^{(t)}(\mathbf{Z}) \geq 0$ *for $\mathbf{Z} \in \mathcal{H}_\phi$;*

(c) $\textbf{logit}_{5,j} = O(1/d)$ *whenver $\mathcal{H}_\phi^\dagger$ happens.*

*Proof.* lemma E.1a is from both the initialization of weights Fact E.3 and Induction E.2a. lemma E.1b is from lemma E.1a and Induction E.1a. lemma E.1c is due to both Induction E.1b and Induction E.1b. $\square$

### E.2.1 Competition between feature combinations.

In this phase we compare the growth of different features using a proxy.

**Lemma E.2** (Approximating gradient with proxy). *Assume Induction E.1. For a fixed $\psi = (j, r, \phi) \in \Sigma^\star$, define the auxiliary sequence*

$$\widetilde{V}_\psi^{(0)} := V_\psi^{(0)}, \qquad \widetilde{V}_\psi^{(t+1)} := \widetilde{V}_\psi^{(t)} + \eta \, \mathbb{E}\left[\textbf{sReLU}'(\widetilde{\Lambda}_{5,j,r}^{(t)}) \, \mathbb{1}_{\mathcal{H}_\phi}\right],$$

*where in $\widetilde{\Lambda}_{5,j,r}^{(t)}$ the weight $V_\psi$ is replaced by $\widetilde{V}_\psi$ (all other weights are unchanged). Then for any feature $\psi \succ \psi_i$ or $\psi \in \Sigma_i \setminus \{\psi_i\}$ and all $t \leq T_\psi^1$, $|V_\psi^{(t)} - \widetilde{V}_\psi^{(t)}| \leq O(\widetilde{V}_\psi^{(t)}/d^{1/2-3c_1})$.*

*Proof.* We need to prove the result by induction. Firstly we notice that the newly defined proxy sequence $\widetilde{V}_\psi^{(t)}$ is monotonically increasing. Moreover, as $\widetilde{V}_\psi^{(t)}$ is initialized the same as $V_\psi^{(t)}$, we can use lemmas E.11 and E.12 to argue its growth by taking

$$x_t = \widetilde{V}_\psi^{(t)} + b_{i,j,r}, \quad \xi = \sum_{p=3,4} \langle \mathbf{W}_{5,j,r,p}, e_{x_0} \rangle + \langle \mathbf{W}_{5,j,r,1}, e_{x_1} \rangle, \quad x_{t+1} \geq x_t + \eta \frac{1}{n_y^2} \mathbb{E}[(x_t + \mu)^{q-1}]$$

And lemmas E.12 and E.13 guarantee $\widetilde{V}_\psi^{(t)} \geq d^{c_1}\mu$ after $\widetilde{O}(\frac{1}{\eta\mu^{q-2}})$ steps. Now assume the bound of difference between $V_\psi^{(t)}$ and $\widetilde{V}_\psi^{(t)}$ holds at a $t \leq T_\psi^1$. Since $t \leq T_\psi^1$, we have $V_\psi^{(t)} \leq d^{c_1}\sigma_0$ by definition of $T_\psi^1$. By Fact E.1, the exact update of $V_\psi$ is

$$V_\psi^{(t+1)} = V_\psi^{(t)} + \eta\Big(\mathbb{E}[(1 - \mathbf{logit}_{5,j}^{(t)})\mathbf{sReLU}'(\Lambda_{5,j,r}^{(t)})\mathbb{1}_{\mathcal{H}_\phi}] - \tfrac{1}{2}\,\mathbb{E}[\mathbf{logit}_{5,j}^{(t)}\mathbf{sReLU}'(\Lambda_{5,j,r}^{(t)})\mathbb{1}_{\mathcal{H}_\phi^\dagger}]\Big).$$

Compare the above with the definition of $\widetilde{V}_\psi^{(t)}$, we first we notice that $\widetilde{V}_\psi^{(t)}$ is dominating $V_\psi^{(t)}$ for all iterations $t \leq T_\psi^1$. Then, we can bound the difference between the two sequences by

$$\big|\widetilde{V}_\psi^{(t+1)} - V_\psi^{(t+1)}\big|$$
$$\leq \big|\widetilde{V}_\psi^{(t)} - V_\psi^{(t)}\big| + \eta\Big|\mathbb{E}\big[\mathbf{sReLU}'(\widetilde{\Lambda}_{5,j,r}^{(t)}) - \mathbf{sReLU}'(\Lambda_{5,j,r}^{(t)})\mathbb{1}_{\mathcal{H}_\phi}\big]\Big|$$
$$+ \eta\Big|\mathbb{E}\big[\mathbf{logit}_{5,j}^{(t)}\mathbf{sReLU}'(\Lambda_{5,j,r}^{(t)})\mathbb{1}_{\mathcal{H}_\phi}\big]\Big| + \frac{\eta}{2}\Big|\mathbb{E}\big[\mathbf{logit}_{5,j}^{(t)}\mathbf{sReLU}'(\Lambda_{5,j,r}^{(t)})\mathbb{1}_{\mathcal{H}_\phi^\dagger}\big]\Big|$$

We bound the three error terms for $t \leq T_\psi^1$:

A. Activation perturbation: by the smoothness of $\mathbf{sReLU}'$ on $[0, \varrho]$ and that both pre-activations are in this range during Phase I, we have

$$\Big|\mathbb{E}[(\mathbf{sReLU}'(\widetilde{\Lambda}_{5,j,r}^{(t)}) - \mathbf{sReLU}'(\Lambda_{5,j,r}^{(t)}))\mathbb{1}_{\mathcal{H}_\phi}]\Big| \leq |\widetilde{V}_\psi^{(t)} - V_\psi^{(t)}|\mathbb{E}[(\widetilde{\Lambda}_{5,j,r}^{(t)})^{q-2}\mathbb{1}_{\mathcal{H}_\phi}]$$

where we have used the fact that $\widetilde{\Lambda}_{5,j,r} \geq \Lambda_{5,j,r}$ because of the dominance of $\widetilde{V}_\psi$.

B. Outside a set of iterations $\mathcal{A}_\phi = \{t \leq T_\psi^1 \mid \mathbf{logit}_{5,j}^{(t)} \geq \frac{1}{\sqrt{d}}$ conditioned on $\mathcal{H}_\phi\}$ whose cardinality is no more than $\widetilde{O}(d^{\frac{1}{2}+c_1}/\eta)$ by Induction E.2a, we have $\mathbf{logit}_{5,j}^{(t)} \leq d^{-1/2}$. Hence for $t \notin \mathcal{A}_\phi$

$$\Big|\mathbb{E}[\mathbf{logit}_{5,j}^{(t)}\mathbf{sReLU}'(\Lambda_{5,j,r}^{(t)})\mathbb{1}_{\mathcal{H}_\phi}]\Big| \leq O(d^{-1/2})\mathbb{E}[\mathbf{sReLU}'(\widetilde{\Lambda}_{5,j,r}^{(t)})\mathbb{1}_{\mathcal{H}_\phi}]$$

Now we can sum over the iterations to get

$$\Bigg(\sum_{s \leq t, s \in \mathcal{A}_\phi} + \sum_{s \leq t, s \notin \mathcal{A}_\phi}\Bigg)\eta\Big|\mathbb{E}\big[\mathbf{logit}_{5,j}^{(s)}\mathbf{sReLU}'(\Lambda_{5,j,r}^{(s)})\mathbb{1}_{\mathcal{H}_\phi}\big]\Big|$$
$$\leq O(\frac{d^{\frac{1}{2}+c_1}}{\eta}) \cdot \eta(d^{c_1}\mu)^{q-1} + O(d^{-1/2})(\widetilde{V}_\psi^{(t)} - \widetilde{V}_\psi^{(0)})$$
$$\leq O(\mu/d^{1/2-c_1})$$

C. Confusing events: before $V_\psi^{(t)} \geq \frac{1}{2}\log d$, we have $\mathbf{logit}_{5,j} = O(1/d)$ in Phase I by lemma E.1. So we get

$$\sum_{s \leq t}\eta\mathbb{E}[\mathbf{logit}_{5,j}^{(t)}|\mathbf{sReLU}'(\Lambda_{5,j,r}^{(t)})|\mathbb{1}_{\mathcal{H}_\phi^\dagger}] \leq O(\frac{1}{\eta\mu^{q-2}})\eta\Big(\frac{(d^{c_1}\mu)^{q-1}}{d}\Big) \ll O(\mu/\sqrt{d})$$

So we have that the accumulated errors from the last two terms are smaller than $O(\mu/d^{1/2-c_1})$. This allow us to bound the difference by using the more naive approach: we can assume the difference

started with $O(\mu/d^{1/2-c_1})$ and now we only need to bound the following sequence for $t \leq T^1_\psi$

$$\delta_{t+1} = \delta_t + O(\eta\delta_t)\mathbb{E}\left[\frac{1}{\widetilde{\Lambda}^{(t)}_{5,j,r}}\mathbf{sReLU}'(\widetilde{\Lambda}^{(t)}_{5,j,r})\mathbb{1}_{\mathcal{H}_\phi}\right], \quad \delta_0 \leq O(\mu/d^{1/2-c_1})$$

Now let $\delta' > \eta$ be some small parameter, we can do some slicing of time $t_0, t_1, \ldots, t_n$ where $t_i = \min\{t \geq 0 \mid \widetilde{V}^{(t)}_\psi \geq (1+\delta')^i\mu\}$ and $(1+\delta')^n\mu \leq \widetilde{V}^{(t+1)}_\psi \leq (1+\delta')^{n+1}\mu$. This produce the following bound:

$$\log\delta_{t+1} - \log\delta_0$$

$$= \sum_{1\leq i\leq n}\sum_{s\in[t_i,t_{i-1}]}\log\left(1 + O(\eta)\mathbb{E}\left[\frac{1}{\widetilde{\Lambda}^{(s)}_{5,j,r}}\mathbf{sReLU}'(\widetilde{\Lambda}^{(s)}_{5,j,r})\mathbb{1}_{\mathcal{H}_\phi}\right]\right)$$

$$\leq \sum_{1\leq i\leq n}\sum_{s\in[t_i,t_{i-1}]}O(\eta)\mathbb{E}\left[\frac{1}{\widetilde{\Lambda}^{(s)}_{5,j,r}}\mathbf{sReLU}'(\widetilde{\Lambda}^{(s)}_{5,j,r})\mathbb{1}_{\mathcal{H}_\phi}\right] \qquad (\log(1+x) \leq x \text{ when } x > 0)$$

$$\leq \sum_{1\leq i\leq n}O\left(\frac{1}{\min_{s\in[t_i,t_{i-1}],\mathbf{Z}\in\mathcal{H}_\phi}\widetilde{\Lambda}^{(s)}_{5,j,r}(\mathbf{Z})}\right)\sum_{s\in[t_i,t_{i-1}]}\eta\left|\mathbb{E}[\mathbf{sReLU}'(\widetilde{\Lambda}^{(s)}_{5,j,r})\mathbb{1}_{\mathcal{H}_\phi}]\right|$$

$$\leq \sum_{1\leq i\leq n}O\left(\frac{1}{(1+\delta')^i\mu}\right)(\widetilde{V}^{(t_{i+1})}_\psi - \widetilde{V}^{(t_i)}_\psi)$$

$$\leq \frac{\delta'\log(\widetilde{V}^{(t)}_\psi/\mu)}{\log(1+\delta')}$$

$$\leq 2\log(\widetilde{V}^{(t)}_\psi/\mu) \qquad (\log(1+\delta')^{\frac{1}{\delta'}} < 2 \text{ if } \delta' \text{ is small enough})$$

So when $\widetilde{V}^{(t)}_\psi \in [d^{c_1}\mu, 2d^{c_1}\mu]$, it holds that $\delta_t \lesssim d^{2c_1}\delta_0 \lesssim \mu/d^{1/2-3c_1}$. This proves the claim. $\qquad\square$

**Lemma E.3** (competition). *Assuming Induction E.1, and let $\psi \in \Sigma^\star$, we shall have for any feature $\psi' \succ \psi \in \Sigma^\star$ or $\psi' \in \Sigma_\psi \setminus \{\psi\}$ that $V^{(T^1_\psi)}_{\psi'} \leq \widetilde{O}(\sigma_0)$.*

*Proof.* We can approximate the sequence $V_{\psi_i}$ using $\widetilde{V}^{(t)}_{\psi_i}$ and straightforwardly use lemma E.12 to compare $\psi_i$ with any $\psi' \succ \psi_i \in \Sigma^\star$. In fact, let's compared the time they reach $V^{(t)}_\psi \geq d^{c_1}\sigma_0$. By lemma E.2, we can just compare $\widetilde{V}^{(t)}_\psi$ before $t = T^1_\psi$, now since by Fact E.3 we know $\widetilde{V}^{(0)}_\psi \geq \widetilde{V}^{(0)}_{\psi'} + \gamma$, we can use lemma E.12 to get that the first time when $\widetilde{V}^{(t)}_{\psi_i} \geq d^{c_1}\sigma_0$, we have $\widetilde{V}^{(t)}_{\psi'} = \widetilde{O}(\sigma_0)$. This combined with lemma E.2 concludes the result. Moreover, we can also obtain bounds on all $\psi \in \Psi$ by discussing different $\psi$:

- For all $\phi' \in \Phi^\star$ such that there exists $\psi' = (j', r', \phi') \prec \psi_i$, $\phi'$ is learned. That means any $\psi$ contains $\phi'$) has gradient too small because of E.2c and therefore $\mathbb{E}[\mathcal{E}_{5,j}\mathbb{1}_{\mathcal{H}_{\phi'}}] \leq \widetilde{O}(\lambda|\Gamma^{(t)}_\psi|)$ and therefore wouldn't be learned.
- All the rest of $\phi' \in \Phi^\star$ has their corresponding $\psi' \succ \psi \in \Sigma^\star$ and therefore lost in the competition and has small growth.
- All other $\psi \in \Psi \setminus \cup_{i'\leq i}\Sigma_{i'}$ lost the competition as well for the same reason.

This concludes the proof. $\qquad\square$

*Proof of Inductions E.1 and E.2 in Phase I.* Let $\psi_i \in \Sigma^\star$. Assume Induction E.1 holds for some $t \leq T^3_{\psi_{i-1}}$, then we shall prove that it continues to hold for $t \in [T^3_{\psi_{i-1}}, T^1_{\psi_i}]$.

- Induction E.1a: This is simple as all $V_\psi$ for $\psi \succ \psi_i$ has updates approximated by $\widetilde{V}^{(t)}_\psi$, which is monotonically increasing, and has minimal update by lemma E.2.
- Induction E.1b: We proved in lemma E.3 that whenever $\psi \prec \psi' \in \Sigma^\star$, $T^1_\psi \leq T^1_{\psi'}$. This does not violate Induction E.1b so the non-overlapping is preserved.

- Induction E.2a: This is proven in lemma E.3.
- Induction E.2b-c: They will be proven when we analyze the end phase of feature learning. They are the corollaries of the end phase guarantee of the previous features.
- Induction E.2c:
- Induction E.1e: We did not violate Induction E.1e either.
- Induction E.2a-d will be proved in Phase III where we argue the end phase of each feature learned.

$\square$

## E.3 Phase II: Feature Growth and Cancellations

Below we present the induction hypothesis for phase II.

**Induction E.3** (Phase II). *For all $\psi = (j, r, \phi) \in \Sigma^\star$ and $t \in [T_\psi^1, T_\psi^2]$, the following holds*

(a) *For $t \in [T_\psi^1, T_\psi^{2,1}]$, $\mathcal{E}_{5,j}^{(t)} \geq 1 - \frac{1}{\sqrt{d}}$ conditioned on $\mathcal{H}_\phi$;*

(b) *For $t \leq T_\psi^{2,1}$, for all $j' \neq j$, we have $\mathbf{logit}_{5,j'}^{(t)} \leq O(\sqrt{d}) \cdot \mathbf{logit}_{5,j'}^{(T_\psi^1)}$.*

(c) *For any $\phi' = (g', y') \neq \phi$, $V_{j,r}^{(t)}(g'), V_{j,r}^{(t)}(y') \in (-B - O(1), \widetilde{O}(\mu))$;*

(d) *For $t \leq T_\psi^{2,2}$, the number of iterations where $V_\psi^{(t+1)} - V_\psi^{(t)} < 0$ is bounded by $\widetilde{O}(1/d^{c_1}\eta)$.*

### E.3.1 Technical Lemmas

**Lemma E.4** (gradient estimation). *Assuming Inductions E.1 and E.3, let $\psi = (j, r, \phi) \in \Sigma^\star, \phi = (g, y)$ and $t \in [T_\psi^{2,1}, T_\psi^2]$, then*

(a) *At any $t \geq T_\psi^{2,1}$, if $V_\psi^{(t)} \in (\frac{1}{3}\log d, \frac{2}{3}\log d)$, then $\Gamma_{j,r,\phi}^{(t)} \geq \Omega(1)$.*

(b) *Any $g' \neq g \in \mathcal{G}, y' \neq y \in \mathcal{Y}$ has the following gradient approximation*

$$\Gamma_{j,r,g'}^{(t)} = -\mathbb{E}\Big[\mathbf{logit}_{5,j}^{(t)}\mathbf{sReLU}'(\Lambda_{5,j,r}^{(t)})\mathbb{1}_{\mathcal{H}_{(g',y)}}\Big] \pm O(\varpi/n_y)$$

$$\Gamma_{j,r,y'}^{(t)} = -\mathbb{E}\Big[\mathbf{logit}_{5,j}^{(t)}\mathbf{sReLU}'(\Lambda_{5,j,r}^{(t)})\mathbb{1}_{\mathcal{H}_{(g,y')}}\Big] \pm O(\varpi/n_y)$$

(c) *let $\phi' = (g', y') \neq \phi \in \Phi_j^\star$ if $V_{j,r}(g, y) \geq B - o(1)$ and $V_{j,r}(g', y') < -B - 2\mu$, we have*

$$\Gamma_{j,r,g'}^{(t)} = \sum_{\widetilde{y} \in \mathcal{Y}, V_{j,r}(g',\widetilde{y}) \in (-B, -\varpi-\mu)} \Omega(\frac{\lambda\varpi}{dn_y^3}) - \mathbb{E}\Big[\mathbf{logit}_{5,j}^{(t)}\mathbf{sReLU}'(\Lambda_{5,j,r}^{(t)})\mathbb{1}_{\mathcal{H}_{(g',y)}}\Big]$$

*and similarly for $\Gamma_{j,r,y'}^{(t)}$.*

*Proof.* Firstly (a) is simple to prove, because when $V_\psi^{(t)} \in (\frac{1}{3}\log d, \frac{2}{3}\log d)$, we have $\mathbf{logit}_{5,j}\mathbb{1}_{\mathcal{H}_\phi^\dagger} \leq O(\frac{1}{d^{c_1}})$, which is from the calculation $F_{5,j} \leq \frac{2}{3}\log d + O(1)$ guaranteed by Inductions E.2 and E.3. For (b), we can estimate the gradient update of $V_{j,r}^{(t)}(g')$ where $g' \neq g$. In fact

$$-\nabla_{V_{j,r}(g')}\mathsf{Loss}^{(t)} = \Gamma_{j,r,g'}^{(t)} = \Gamma_{j,r,g'}^{+,(t)} - \Gamma_{j,r,g'}^{-,(t)}$$

$$= \Gamma_{j,r,g'}^{+,(t)} - \sum_{y' \in \mathcal{Y}} \mathbb{E}\Big[\mathbf{logit}_{5,j}^{(t)}\mathbf{sReLU}'(\Lambda_{5,j,r}^{(t)})\mathbb{1}_{\mathcal{H}_{(g',y')}}\Big]$$

$$= -\mathbb{E}\Big[\mathbf{logit}_{5,j}^{(t)}\mathbf{sReLU}'(\Lambda_{5,j,r}^{(t)})\mathbb{1}_{\mathcal{H}_{(g',y)}}\Big] \pm \max\{\widetilde{O}(\mu^{q-1}), \varpi\}$$

where the last one is because for $y' \neq y$, the activation gradient $\mathbf{sReLU}'(\Lambda_{5,j,r}^{(t)})$ is either positive and bounded by $\widetilde{O}(\mu^{q-1})$ or negative and bounded by $\varpi$ while $\mathbf{logit}_{5,j}\mathbb{1}_{\mathcal{H}_{\phi'}} \leq 1$. Similarly we can also get the same for $V_{j,r}(y')$ where $y' \neq y$:

$$-\nabla_{V_{j,r}(y')}\mathsf{Loss}^{(t)} = \Gamma_{j,r,y'}^{(t)} = -\mathbb{E}\Big[\mathbf{logit}_{5,j}^{(t)}\mathbf{sReLU}'(\Lambda_{5,j,r}^{(t)})\mathbb{1}_{\mathcal{H}_{(g',y)}}\Big] \pm O(\varpi/n_y^2)$$

To prove (c), notice that when $V_{j,r}(g', y') < -B - 2\mu$, the part of $\Gamma_{j,r,g'}^{+,(t)} = 0$ due to $\Lambda_{5,j,r}$ exceeding the boundary $-B$. We can group different $\widetilde{y} \neq y \in \mathcal{Y}$ by their feature magnitude and obtain the lower bound. The factor of $\frac{\lambda}{dn_y}$ is due to the logit lower bound from Fact E.2. $\qquad\square$

**Lemma E.5** (Feature Magnitude). *Assume Inductions E.1 and E.3 holds, then for any $\psi \in \Sigma^\star$,*

(a) *at $t = T_\psi^{2,1}$, we have $V_{j,r}^{(t)}(g) = \frac{1}{4}\log d \pm o(1)$, $V_{j,r}^{(t)}(y) = \frac{1}{4}\log d \pm o(1)$;*

(b) *when $\mathcal{C}_\psi^+(\delta)$ hold for some $\delta \leq O(\lambda/d^{c_1})$ at some $t \geq T_\psi^{2,1}$, we have $V_\psi^{(t)} \geq B - O(d^{c_1}\mu)$, that is, $\mathcal{F}_{\psi,1}(d^{c_1}\mu)$ holds.*

*Proof.* Let's proceed the proof one by one.

- Proof of (a) :This can be computed by comparing the gradient of $V_{j,r}^{(t)}(g)$ which is $\Gamma_{j,r,g}^{(t)} = \Gamma_{j,r,g}^{+,(t)} - \Gamma_{j,r,g}^{-,(t)}$ and the gradient of $V_{j,r}^{(t)}(y)$ which is $\Gamma_{j,r,g}^{(t)} = \Gamma_{j,r,y}^{+,(t)} - \Gamma_{j,r,y}^{-,(t)}$. Now since

$$\Gamma_{j,r,g}^{+,(t)} = \Gamma_{j,r,y}^{+,(t)} \geq (1 - O(\frac{1}{\sqrt{d}}))\mathbf{sReLU}'(V_{j,r}^{(t)}(g) + V_{j,r}^{(t)}(y) + O(\mu))$$

and that

$$\Gamma_{j,r,g}^{-,(t)} = O(\frac{1}{\sqrt{d}}\mathbf{sReLU}'(V_{j,r}^{(t)}(g) + O(\mu))|), \quad \Gamma_{j,r,y}^{-,(t)} = O(\frac{1}{\sqrt{d}}\mathbf{sReLU}'(V_{j,r}^{(t)}(y) + O(\mu))|)$$

one could see that $\sum_{t \leq T_\psi^{2,1}} |\Gamma_{j,r,g}^{(t)} - \Gamma_{j,r,y}^{(t)}| \leq \widetilde{O}(\frac{1}{\sqrt{d}})$ and thus (a) holds.

- Proof of (b): from lemma E.4a we know that when $\mathcal{C}_\psi^+(\delta)$ holds for some small $\delta = O(\frac{\lambda}{d^{c_1}})$ it is not because $V_\psi^{(t)}$ dropped below $\frac{1}{2}\log d$, thus when Inductions E.1 and E.3 holds, it can only be because $V_\psi^{(t)} \geq B - O(d^{c_1}\mu)$, otherwise

$$\Gamma_\psi^{+,(t)} = \mathbb{E}[(1 - \mathbf{logit}_{5,j}^{(t)})\mathbb{1}_{\mathcal{H}_\phi}] \geq \lambda\mathbf{Pr}(\mathcal{H}_\phi) \gg \frac{\lambda}{d^{c_1}} \qquad (\text{if } V_\psi^{(t)} \leq B - O(d^{c_1}\mu))$$

which concludes the proof.

$\qquad\square$

**Lemma E.6** (Gradient Stationarity). *Let $\psi \in \Sigma^\star$, assume Inductions E.1 and E.2 holds at $t \in [T_\psi^1, T_\psi^2]$, and choose $\delta \gg n_y\varpi$, then we have the following guarantees:*

(a) *Define $\mathcal{B}_\delta^+ = \{t \in [T_\psi^{2,1}, T_\psi^2] \mid \mathcal{C}_\psi^+(\delta) \text{ doesn't hold}\}$, then $|\mathcal{B}_\delta^+| \leq O(\frac{n_y \log^2 d}{\eta\delta})$;*

(b) *Define $\mathcal{B}_\delta^- = \{t \in [T_\psi^{2,1}, T_\psi^2] \mid \mathcal{C}_\psi^-(\delta) \text{ doesn't hold}\}|$, then $|\mathcal{B}_\delta^-| \leq O(\frac{n_y \log^3 d}{\eta\delta})$;*

(c) *Finally, at $t = T_\psi^2$, the feature shape $\mathcal{F}_\psi(\delta_1, \delta_2)$ holds with $\delta_1 = d^{c_1}\mu$ and $\delta_2 = (\frac{n_y^3 d\varpi}{\lambda})^{\frac{1}{q-1}}$.*

*Proof.* Let us start with (a). First by writing $\psi = (j, r, \phi)$, we define the following quantity

$$\Upsilon^{(t)} := V_{j,r}^{(t)}(\phi) - \sum_{\phi' \in \Phi_j^\star \setminus \{\phi\}} V_{j,r}^{(t)}(\phi') \tag{33}$$

which sits at the central of our proof of counter-argument.Now we have for any $T \geq T_\psi^{2,1}$

$$\Upsilon^{(t)} = V_{j,r}^{(T)}(\phi) - \sum_{\phi' \in \Phi_j^\star \setminus \{\phi\}} V_{j,r}^{(T)}(\phi')$$

$$= V_{j,r}^{(T_\psi^{2,1})}(\phi) - \sum_{\phi' \in \Phi_j^\star \setminus \{\phi\}} V_{j,r}^{(T_\psi^{2,1})}(\phi') + \sum_{s \in [T_\psi^{2,1}, t)} \eta\Gamma_{j,r,\phi}^{(t)} - \sum_{\phi' \in \Phi_j^\star \setminus \{\phi\}} \sum_{s \in [T_\psi^{2,1}, t)} \eta\Gamma_{j,r,\phi'}^{(t)}$$

We can further rewrite the $\Gamma$s on the RHS to

$$\sum_{t\in[T_\psi^{2,1},T)} \eta\left(\Gamma_{j,r,\phi}^{(t)} - \sum_{\phi'\in\Phi_j^\star\setminus\{\phi\}} \eta\Gamma_{j,r,\phi'}^{(t)}\right)$$

$$= \sum_{t\in[T_\psi^{2,1},T)} \eta\left(\Gamma_{j,r,\phi}^{+,(t)} - \Gamma_{j,r,\phi}^{-,(t)} - \sum_{\phi'\neq\phi\in\Phi_j^\star} \Gamma_{j,r,\phi'}^{+,(t)} + \sum_{\phi'\neq\phi\in\Phi_j^\star} \Gamma_{j,r,\phi'}^{-,(t)}\right) \qquad (34)$$

Now we can start to estimate the RHS. Define $P_1 = \{\phi' \neq \phi \in \Phi^\star \mid \phi' \text{ learned}\}$[7] and $P_2 = \{\phi' \neq \phi \in \Phi^\star \mid \phi' \text{ not learned}\}$. The sum $\sum_{\phi'\neq\phi\in\Phi_j^\star} \Gamma_{j,r,\phi'}^{+,(t)}$ can be decomposed and bounded by

$$-O(\eta n_y \varpi) \leq \sum_{\phi'\in P_1} \eta\Gamma_{j,r,\phi}^{+,(t)} + \sum_{\phi'\in P_2} \eta\Gamma_{j,r,\phi}^{+,(t)} \leq \widetilde{O}(\eta\lambda\mu^{q-2})$$

and then by the same methods of bounding the gradients, we have that

$$\sum_{\phi'\neq\phi\in\Phi_j^\star} \Gamma_{j,r,\phi'}^{-,(t)}$$

$$= \sum_{\phi'=(g',y')\in\Phi_j^\star\setminus\{\phi\}} \mathbb{E}\left[\mathbf{logit}_{5,j}^{(t)}\mathbf{sReLU}'(\Lambda_{5,j,r}^{(t)})\Big(\sum_{g''\neq g'\in\mathcal{G}} \mathbb{1}_{\mathcal{H}_{(g'',y')}} + \sum_{y''\neq y'\in\mathcal{Y}} \mathbb{1}_{\mathcal{H}_{(g',y'')}}\Big)\right]$$

$$= \sum_{g'\neq g} \mathbb{E}[\mathbf{logit}_{5,j}^{(t)}\mathbf{sReLU}'(\Lambda_{5,j,r}^{(t)})\mathbb{1}_{\mathcal{H}_{(g',y)}}] + \sum_{y'\neq y} \mathbb{E}[\mathbf{logit}_{5,j}^{(t)}\mathbf{sReLU}'(\Lambda_{5,j,r}^{(t)})\mathbb{1}_{\mathcal{H}_{(g,y')}}] \pm O(\varpi)$$

$$= \Gamma_{j,r,\phi}^{-,(t)} \pm O(\varpi)$$

where ① is because for any incorrect feature combination $\phi' = (g',y') \notin \Phi_j^\star$ and does not share a component with $\phi$, the activation of $\Lambda_{5,j,r}$ conditioned on $\mathcal{H}_{\phi'}$ is within $[-B, \widetilde{O}(\mu^{q-1})]$ and thus the negative gradient is upper bounded by $\varpi/n_y^2$ and lower bounded by $-\widetilde{O}(\mu^{q-1})$. Inserting this back to Eq. (34), we shall have

$$\sum_{t\in[T_\psi^{2,1},T)} \eta\left(\Gamma_{j,r,\phi}^{(t)} - \sum_{\phi'\in\Phi_j^\star\setminus\{\phi\}} \eta\Gamma_{j,r,\phi'}^{(t)}\right) = \sum_{t\in[T_\psi^{2,1},T)} \eta\Gamma_{j,r,\phi}^{+,(t)} \pm O((T-T_\psi^{2,1})\eta\varpi)$$

Now we have arrived at the desired estimate of the update, finally, we have the following counter-argument: if $\mathcal{C}_\psi^+(\delta)$ does not hold for $\Omega(\frac{n_y \log^2 d}{\eta\delta})$ iterations, letting $T-1$ be the last iteration which $\mathcal{C}_\psi^+(\delta)$ does not hold, we have

$$\Upsilon^{(T)} \geq \sum_{t\in[T_\psi^{2,1},T)} \eta\Gamma_{j,r,\phi}^{+,(t)}$$

$$\geq \sum_{t\in[T_\psi^{2,1},T]\setminus B_\delta^+} \eta\delta + \Omega((T-T_\psi^{2,1})\eta\varpi) \qquad (\text{For } t\in B_\delta^+, \Gamma_{j,r,\phi}^{+,(t)} \geq 0)$$

$$\geq \sum_{t\in[T_\psi^{2,1},T]\setminus B_\delta^+} \eta\delta - O(\frac{n_y \log^2 d}{\eta\delta}\eta\varpi)$$

$$\geq \Omega(\frac{n_y \log^2 d}{\eta\delta})\eta\delta \qquad (\text{by choosing } \delta \gg \varpi)$$

$$\geq \Omega(n_y \log^2 d)$$

---

[7] By $\phi'$ is learned we mean there is a feature $\psi' \prec \psi \in \Sigma^\star$ such that $\psi' = (j,r,\phi')$.

which is impossible because the gradient would have vanished when $\Upsilon^{(T)} \geq n_y \log^{1.5} d$ as that would mean $\exists \phi \in \Phi_j^\star$ such that $|V_{j,r}^{(t)}(\phi)| \geq \log^2 d$ and it is forbidden in our setting, which proves the claim via contradiction.

**Proof of lemma E.6b**. Actually, by applying (a) combined with lemma E.5b, we know that for some $T \geq T_\psi^{2,1} + \widetilde{\Omega}(\frac{1}{\eta\lambda})$ it holds $V_\psi^{(T)} \geq B - O(d^{c_1}\mu)$. From this iteration forward we shall prove that $V_\psi^{(t)}$ stays above $B - O(d^{c_1}\mu)$ and can help. In fact, at $t = T$, $V_{j,r}^{(t)}(V) \in (-O(B), \widetilde{O}(\mu))$ for $v \in \mathcal{G} \cup \mathcal{Y} \setminus \{g, y\}$ by Induction E.3. Say $v = g' \neq g$, then due to lemma E.4, its gradient is approximated by

$$-\mathbb{E}\Big[\mathbf{logit}_{5,j}^{(t)}\mathbf{sReLU}'(\Lambda_{5,j,r}^{(t)})\mathbb{1}_{\mathcal{H}_{(g',y)}}\Big] \pm O(\varpi/n_y) \tag{35}$$

Let $\delta \geq \varpi$, we will find an iteration $t \leq T_\psi^{2,1} + O(\frac{n_y \log^3 d}{\eta\delta})$ that satisfy $\mathcal{C}_\psi^-(\delta)$. By applying (a), we first get that $\mathcal{C}_\psi^+(\delta)$ holds for no more than $O(\frac{n_y \log^2 d}{\eta\delta})$ iters after $T_\psi^{2,1}$ and before $T_\psi^2$.

Let's assume this is the case, that $\mathcal{C}_\psi^-(\delta)$ doesn't hold for more than $\Omega(\frac{n_y \log^3 d}{\eta\delta})$ iterations. In this case, it must be that the gradient $\Gamma_{j,r,\phi}^{-,(t)} < -\delta$ is negative, since when the gradient is positive it has an upper bound $O(\varpi/n_y) = o(\delta)$ and therefore cannot violate the condition $\mathcal{C}_\psi^+(\delta)$. Therefore the updates $\sum_{s \leq t} \Gamma_{j,r,\phi}^{-,(s)}$ must accumulate to lower than

$$V_{j,r}^{(t)}(\phi) \leq \Upsilon^{(t)} - \Omega(\frac{n_y \log^3 d}{\eta\delta}) \times \eta\delta = -\Omega(n_y \log^3 d)$$

which is impossible in our setting. This proved the result for any $\delta \gg n_y\varpi$.

**Proof of lemma E.6c**. Let $\delta = n_y^2\varpi$, which is covered by lemma E.6a-b, we can find an iteration where $|\Gamma_{j,r,\phi'}^{(t)}| \leq \delta \; \phi' \in \Phi_j^\star \setminus \{\phi\}$. This implies we have found a $t$ such that

$$|\mathbb{E}[\mathbf{logit}_{5,j}^{(t)}\mathbf{sReLU}'(\Lambda_{5,j,r})\mathbb{1}_{\mathcal{H}_{\phi'}}]| \leq n_y^2\varpi \implies |\mathbf{sReLU}'(V_{j,r}^{(t)}(\phi') + O(\mu))| \leq O(\frac{n_y^3 d\varpi}{\lambda})$$
$$\text{(We lower bounded the logit by Fact E.2)}$$

Thus we arrive at $|V_{j,r}(\phi')| \leq O((\frac{n_y^3 d\varpi}{\lambda})^{\frac{1}{q-1}})$ Remember in the proof of b, $V_\psi^{(t)} = \frac{1}{2}(V_{j,r}^{(t)}(g) + V_{j,r}^{(t)}(y)) \geq B - O(d^{c_1}\mu)$ at $t \geq T + \widetilde{\Omega}(\frac{d^{c_1}}{\eta\lambda})$. So we have for any $\phi' = (g', y') \in \Phi_j^\star \setminus \{\phi\}$,

$$\frac{1}{2}(V_{j,r}^{(t)}(g') + V_{j,r}^{(t)}(y')) \leq -\frac{1}{2}(V_{j,r}^{(t)}(g) + V_{j,r}^{(t)}(y)) + O(\frac{1}{d^{1/3q}}) \leq -B + O((\frac{n_y^3 d\varpi}{\lambda})^{\frac{1}{q-1}})$$

Now we have verified that given $\delta = n_y^2\varpi$, there is a iteration $T_\psi^{2,3}$ which gives the near-perfect feature shape $\mathcal{F}_\psi(\delta_1, \delta_2)$ with $\delta_1 = d^{c_1}\mu, \delta_2 = (\frac{n_y^3 d\varpi}{\lambda})^{\frac{1}{q-1}}$. We shall prove that this feature shape will hold until $t = T_2$.

For $t \geq T_\psi^{2,3}$, we verify $\mathcal{F}_\psi(\delta_1, \delta_2)$ in Definition E.4 one by one:

1. $\mathcal{F}_{\psi,1}(d^{c_1}\mu)$: By lemma E.6a we shall find a step after $t \geq T_\psi^{2,1} + O(\frac{d^{c_1}}{\eta\lambda})$ at which $\mathcal{F}_{\psi,1}(d^{c_1}\mu)$ is satisfied. In fact, the majority of the steps after $T_\psi^{2,1} + O(\frac{d^{c_1}}{\eta\lambda})$ will satisfy this. We shall prove that combined with the above feature shape guarantee, $\mathcal{F}_{\psi,1}(d^{c_1}\mu)$ will hold true until $t = T_1$.

2. $\mathcal{F}_{\psi,2}((\frac{n_y^3 d\varpi}{\lambda})^{\frac{1}{q-1}})$: From the above argument that applies lemma E.6b, we shall find a time step where $\mathcal{F}_{\psi,1}(d^{c_1}\mu)$ and $\mathcal{F}_{\psi,2}(d^{-1/3q})$ hold. Now we discuss the possible values of the gradient $\Gamma_{\psi'}^{-,(t)}, \psi' \in \Sigma_\psi^{\dagger,1}$. In fact, let $\phi' = (g', y') \in \Phi_j^\star \setminus \{\phi\}$, we have

    - Regime A: If $V_{j,r}(\phi') < -B - 2\mu$, then there is no negative gradient for $V_{j,r}(g')$ and $V_{j,r}(y')$ unless $V_{j,r}(g', y)$ or $V_{j,r}(g, y')$ is positive. In fact, only one of the feature can be positive to have negative gradient, and in that case the gradient of $\Gamma_{j,r,v}^{(t)}, v \in \{g', y'\}$ is pointing to the convergence direction $|V_{j,r}(g', y)| \to 0$.

– Regime B: If $V_{j,r}(\phi') \geq -B - 2\mu$, then there is a possibility of negative gradient for $V_{j,r}(g')$ and $V_{j,r}(y')$ even when $V_{j,r}(g', y)$ or $V_{j,r}(g, y')$ is negative, In this case the feature $V_{j,r}(g', y)$ or $V_{j,r}(g, y')$ might not go the desired direction to converge to zero, but given a $\delta > n_y^2 \varpi$, $\Gamma^{-,(t)}_{j,r,(g',y)}$ or $\Gamma^{-,(t)}_{j,r,(g,y')}$ cannot stay above $\delta$, otherwise the term will dominate and revert the gradient direction. On the other hand, when one side has $V_{j,r}(g, y') < -\Omega((\frac{n_y^3 d\varpi}{\lambda})^{\frac{1}{q-1}})$, we have

$$V_{j,r}(\phi') \leq -V_{j,r}(\phi) - \Omega((\frac{n_y^3 d\varpi}{\lambda})^{\frac{1}{q-1}}) \leq -B - 2\mu$$

which is back in regime A and therefore will converge back.

Induction over the above two regimes for all $t \in [T_\psi^{2,3}, T_\psi^2]$ showed that the feature will indeed satisfy $\mathcal{F}_{\psi,2}(\delta)$ for all $\delta > n_y^2 \varpi$ at $t$.

3. $\mathcal{F}_{\psi,3}((\frac{n_y^3 d\varpi}{\lambda})^{\frac{1}{q-1}})$: We need to use both $\mathcal{F}_{\psi,1}$ and $\mathcal{F}_{\psi,2}$ to get the desired result. In fact, we have for any $\psi' = (j, r, (g', y')) \in \Sigma^{\dagger,2}$,

$$V_{\psi'}^{(t)} = V_{j,r}^{(t)}(g') + V_{j,r}^{(t)}(y') + V_{j,r}^{(t)}(g) + V_{j,r}^{(t)}(y) - V_{j,r}^{(t)}(\phi) = -B + O((\frac{n_y^3 d\varpi}{\lambda})^{\frac{1}{q-1}})$$

because $V_{j,r}^{(t)}(g) + V_{j,r}^{(t)}(y')$ and $V_{j,r}^{(t)}(g') + V_{j,r}^{(t)}(y)$ are bounded due to $\mathcal{F}_{\psi,2}$.

4. $\mathcal{F}_{\psi,4}(d^{c_1}\mu)$: The number of iterations $[T_\psi^1, T_\psi^2]$ is no more than $\widetilde{O}(\frac{1}{\eta(d^{c_1}\mu)^{q-2}})$, so by using lemma E.13 along with the induction from the result at $T_\psi^1$ shall conclude the proof.

5. $\mathcal{F}_{\psi,5}(((\frac{n_y^3 d\varpi}{\lambda})^{\frac{1}{q-1}}))$: By using lemma E.7 and $\mathcal{F}_{\psi,2}$ after $t \geq T_\psi^{2,3}$, we have the result.

This proved (c). $\qquad\qquad\qquad\qquad\qquad\qquad\qquad\qquad\qquad\qquad\qquad\qquad\qquad\qquad\quad\square$

**Lemma E.7** (Symmetry of Features). *Let $\psi = (j, r, (g, y)) \in \Sigma^\star$ and $t \geq T_\psi^{2,3}$, we have that if $|V_{\psi'}| \leq \delta$ for all $\psi' \in \Sigma^{\dagger,1}$, then*

$$|V_{j,r}^{(t)}(g) - V_{j,r}^{(t)}(y)| \leq O(\delta)$$

*Proof.* For a feature $\psi = (j, r, \phi) \in \Sigma^\star$, $\phi = (g, y)$, their update can be represented as:

$$V_{j,r}^{(t)}(g) = U_{g,y} - \sum_{y' \neq y} R_{g,y'} + V_{j,r}^{(0)}(g)$$

$$V_{j,r}^{(t)}(y) = U_{g,y} - \sum_{g' \neq g} R_{g',y} + V_{j,r}^{(0)}(y)$$

where $U_{g,y} = \sum_{s \leq t} \eta \Gamma_{j,r,\phi}^{+,(s)}$ and $R_{g,y'}$ is

$$R_{g,y'} = \sum_{s \leq t} \eta \mathbb{E}[\mathbf{logit}_{5,j}^{(s)} \mathbf{sReLU}'(\Lambda_{5,j,r}^{(s)}) \mathbb{1}_{\mathcal{H}_{g',y}}]$$

Therefore:

$$\sum_g V_{j,r}^{(t)}(g) = \sum_{\phi \in \Phi_j^\star} U_\phi - \sum_g \sum_{y' \neq y} R_{g,y'} = \sum_{\phi \in \Phi_j^\star} U_\phi - \sum_{\phi^\dagger \in \Phi_j^\dagger} R_{\phi^\dagger} + V_{j,r}^{(0)}(g)$$

$$\sum_y V_{j,r}^{(t)}(y) = \sum_{\phi \in \Phi_j^\star} U_\phi - \sum_y \sum_{g' \neq g} R_{g,y'} = \sum_{\phi \in \Phi_j^\star} U_\phi - \sum_{\phi^\dagger \in \Phi_j^\dagger} R_{\phi^\dagger} + V_{j,r}^{(0)}(y)$$

Thus

$$\sum_{g'} V_{j,r}^{(t)}(g') = \sum_{y'} V_{j,r}^{(t)}(y') \pm O(\mu)$$

Moreover, we have assumed

$$|V_{j,r}^{(t)}(g) + V_{j,r}^{(t)}(y')| = O(\delta), \forall y' \neq y, \quad |V_{j,r}^{(t)}(y) + V_{j,r}^{(t)}(g')| = O(\delta), \forall g' \neq g$$

Thus

$$\sum_{g'} V_{j,r}^{(t)}(g') = V_{j,r}^{(t)}(g) - (n_y - 1)V_{j,r}^{(t)}(y) \pm O(n_y\delta) = V_{j,r}^{(t)}(y) - (n_y - 1)V_{j,r}^{(t)}(g) \pm O(n_y\delta)$$

Therefore

$$V_{j,r}^{(t)}(g) = V_{j,r}^{(t)}(y) + O(\delta) + O(\mu)$$

inserting $\delta = O((\frac{n_y^2 d\varpi}{\lambda})^{\frac{1}{q-1}})$ which is from lemma E.6 proves the claim. □

**Lemma E.8** (Arrival time estimates). *Assuming Inductions E.1 and E.3 for $\psi \in \Sigma^\star$, we have*

(a) $T_\psi^{2,1} - T_\psi^1 \leq \widetilde{O}(1/\eta(d^{c_1}\sigma_0)^{q-2})$;

(b) $T_\psi^{2,2} - T_\psi^{2,1} \leq O(\frac{d^{\frac{1}{2}+c_1}}{\eta})$;

(c) $T_\psi^{2,3} - T_\psi^{2,2} \leq \widetilde{O}(\frac{1}{\eta\varpi})$;

(d) $T_\psi^2 - T_\psi^1 \leq \widetilde{O}(\frac{1}{\eta(d^{2c_1}\mu)^{q-2}})$

*Proof.*     • Proof of (a): Once $V_\psi^{(t)} \geq d^{c_1}\sigma_0$, we can apply lemma E.13 to compute the number of iteration needed to reach $V_\psi^{(t)} \geq \frac{1}{2}\log d$. In fact, by Induction E.3a-b we know that

$$V_\psi^{(t+1)} \geq V_\psi^{(t)} + \eta(1 - O(\frac{1}{\sqrt{d}}))\mathbf{sReLU}'(V_\psi^{(t)} + O(\mu)) \geq V_\psi^{(t)} + \Omega(\eta(V_\psi^{(t)})^{q-1})$$

Since $V_\psi^{(T_\psi^1)} \geq d^{c_1}\sigma_0$, we have by lemma E.13 that $T_\psi^{2,1} - T_\psi^1 \leq O(\frac{1}{\eta(d^{c_1}\sigma_0)^{q-2}})$.

• Proof of (b): Simply apply lemma E.6 with $\delta = \frac{1}{d^{\frac{1}{2}+c_1}}$;

• Proof of (c): Again by applying lemma E.6 with $\delta = n_y^2\varpi$;

• Proof of (d): Follow the definition of $T_\psi^2$.

□

*Proof of Induction E.3.* The proof of time steps are all in lemma E.8 and lemma E.6. We do not repeat here. □

## E.4  Phase III: Convergence

We present the induction hypothesis in this phase.

**Induction E.4** (Phase III, final). *Let $\psi = (j, r, \phi) \in \Sigma^\star$, for $t \in [T_\psi^2, T_1]$, the following holds:*

(a) *At $t \in [T_\psi^2, T_{1,1}]$, we have $\mathcal{F}_\psi(\delta_1, \delta_2)$ holds with $\delta_1 = d^{c_1}\mu$, $\delta_2 = (\frac{n_y^2 d\varpi}{\lambda})^{\frac{1}{q-1}}$;*

(b) *At $t \in [T_\psi^3, T_1]$, we have $\mathcal{F}_\psi(\delta_1, \delta_2)$ holds with $\delta_1 = d^{c_1}\mu$, $\delta_2 = (n_y^2\varpi)^{\frac{1}{q-1}}$.*

We need a lemma to describe the logit distribution when all features are learned.

**Lemma E.9** (Logit shape at convergence). *Assuming Inductions E.1 and E.4. Let $\psi \in \Sigma^\star$ be the last of $\Sigma^\star$, then at $t = T_\psi^2$, the followings hold:*

1. *For $\phi \in \Phi_j^\star$, $\mathbf{logit}_{5,j}^{(t)} \geq 1 - O(\lambda)$ conditioned on $\mathcal{H}_\phi$.*

2. *For $\phi \notin \Phi_j^\star$, $\mathbf{logit}_{5,j}^{(t)} \leq O(\lambda/d)$ conditioned on $\mathcal{H}_\phi$.*

*Proof.* Since $\psi$ is the last feature in the learning curriculum $\Sigma^\star$, all feature at $t = T_\psi^2$ has satisfied $\mathcal{F}_\psi(\delta_1, \delta_2)$ for $\delta_1 = d^{c_1}\mu$, $\delta_2 = (\frac{n_y^2 d\varpi}{\lambda})^{\frac{1}{q-1}}$. Thus we can bound the logit by

$$\mathbf{logit}_{5,j}^{(t)} = \frac{e^{F_{5,j}^{(t)}(\mathbf{Z})}}{e^{F_{5,j}^{(t)}(\mathbf{Z})} + d} = \frac{e^{B-o(1)}}{e^{B-o(1)} + d} = 1 - O(\lambda) \qquad \text{(conditioned on } \mathbf{Z} \in \mathcal{H}_\phi, \phi \in \Phi_j^\star)$$

and

$$\mathbf{logit}_{5,j}^{(t)} = \frac{e^{F_{5,j}^{(t)}(\mathbf{Z})}}{e^{F_{5,j}^{(t)}(\mathbf{Z})} + d} = \frac{e^{o(1)}}{e^{B-o(1)} + d - 1 + e^{o(1)}} = O(\frac{\lambda}{d}) \quad \text{(conditioned on } \mathbf{Z} \in \mathcal{H}_\phi, \phi \in \Phi_j^\star)$$

which is the desired result. $\qquad\qquad\square$

**Lemma E.10** (Gradient Bounds, Phase III). *Assuming Inductions E.1 and E.4 holds, and let $\psi = (j, r, \phi) \in \Sigma^\star$, then at $t \in [T_{1,1}, T_1]$, the followings hold*

(a) $\Gamma_\psi^{+,(t)} \geq \Omega(\lambda)$ *when* $V_\psi^{(t)} \leq B - d^{c_1}\mu$;

(b) $|\Gamma_{j,r,v}^{-,(t)}| \leq O(\frac{\lambda\varpi}{dn_y})$ *for any* $v \in \mathcal{G} \cup \mathcal{Y}$;

(c) *For* $\phi' \in \Phi_j^\star \setminus \{\phi\}$, *we have*

$$\Gamma_{j,r,\phi'}^{+,(t)} = \begin{cases} -\Theta(\lambda\varpi/n_y^2), & \text{if } V_{j,r,\phi'}^{(t)} \in [-B + 2\mu, -\varpi] \\ \in [-\Theta(\lambda\varpi/n_y^2), -\Theta(\lambda\varpi/dn_y^2)] \cup \{0\}, & \text{if } V_{j,r,\phi'}^{(t)} \in [-B - 2\mu, -B + 2\mu] \end{cases}$$

*Proof.* lemma E.10a is simple when lemma E.9a holds. lemma E.10b is basically redoing the calculations in lemma E.4. lemma E.10c is because when $V_{j,r,\phi'}^{(t)} \geq -B + 2\mu$, we have

$$\begin{aligned} \mathbb{E}[\mathcal{E}_{5,j}^{(t)}\mathbf{sReLU}'(\Lambda_{5,j,r})\mathbb{1}_{\mathcal{H}_\phi'}] &= \mathbb{E}[(1 - \mathbf{logit}_{5,j}^{(t)})\mathbf{sReLU}'(\Lambda_{5,j,r}^{(t)})\mathbb{1}_{\mathcal{H}_\phi'}] \\ &\geq \lambda\mathbf{sReLU}'(V_{j,r,\phi'}^{(t)} + \mu \pm o(\mu))\mathbf{Pr}(\mathcal{H}_{\phi'}) \\ &\geq -\Theta(\lambda\varpi/n_y^2) \end{aligned}$$

And the rest is simply the property of $\mathbf{sReLU}'$ at the negative boundary when some activations of $x_0, x_1$ crosses the boundary. $\qquad\qquad\square$

### E.4.1 Proof of Induction in Phase III and theorem E.1

*Proof of Inductions E.1, E.2 and E.4.* We prove the induction by reusing the proof of lemma E.6.

- Proof of Induction E.4a: The same as in the proof of lemma E.6c, with longer time. We do not use any property of $T_\psi^3$ so no additional handling is needed.

- Proof of Induction E.4b: When all feature is learned, that is, after $t = T_\psi^2$ for the last feature $\psi \in \Sigma^\star$, we have perfect feature shape and therefore have much better logit shape as described by lemma E.9. We can reuse the proof of lemma E.6a-b to get the following result: for any $\delta = \omega(\frac{\lambda\varpi}{dn_y})$, we can guarantee that $\mathcal{C}_\psi(\delta)$ doesn't hold for more than $\widetilde{O}(\frac{1}{\eta\delta})$ iterations. And beyond that, we can just reuse the argument in the proof of lemma E.6c to obtain the desired result.

- Proof of Induction E.1a-b: They are correct in Phase III because there is no violation.

- Proof of Induction E.2a-b: They are proven in previous phases.

- Proof of Induction E.2c: For predecessor feature $\psi' \prec \psi \in \Sigma^\star$, we proved in this phase that they maintain feature shape until $T_{\psi'}^3$, and improved their feature at $T_{\psi'}^3$.

- Proof of Induction E.2d: This is proven in previous phases.

$\qquad\qquad\square$

*Proof of theorem E.1.* Since all feature in $\Sigma^\star$ have shape $\mathcal{F}_\psi(\delta_1, \delta_2)$ for $\delta_1 = d^{c_1}\mu$, $\delta_2 = \varpi^{\frac{1}{q-1}}$ at the end of Induction E.4, we have proven theorem E.1. $\qquad\qquad\square$

### E.5 Auxiliary Technical Tools

First we need a Bernstein inequality for U-statistics

**Lemma E.11** (concentration inequality for pseudo-U-statistics). *Let $x_1, \ldots, x_n$ be different symbols, and let $m \ll n$ be such that $n \equiv 0 \pmod{n}$. Suppose for some function $h$ with $|h| \le M$ the random variables $h(x_{i_1}, x_{i_2}, \ldots, x_{i_m})$ and $h(x_{i'_1}, x_{i'_2}, \ldots, x_{i'_m})$ are independent and identically distributed as long as $\{x_{i_1}, x_{i_2}, \ldots, x_{i_m}\} \cap \{x_{i'_1}, x_{i'_2}, \ldots, x_{i'_m}\} = \emptyset$, then the pseudo-U-statistic*

$$U_{m,n} = \frac{1}{\binom{n}{m}} \sum_{0 \le i_1 < i_2 < \cdots < i_m \le n} h(x_{i_1}, x_{i_2}, \ldots, x_{i_m})$$

*satisfies* $\mathbf{Pr}(|U_{n,m} - \mathbb{E}[U_{n,m}]| \ge t) \le e^{-\frac{nt^2}{mM^2}}$

*Proof.* The proof is the same as in [108]. □

We present two lemmas related to the tensor power method.

**Lemma E.12** (TPM in expectation). *Let $p > 3$ be an integer, and $\mu, \sigma > 0$ satisfying $\mu \gg \sigma\sqrt{\log d}$. Now suppose $\xi \sim \mathcal{N}(0, \sigma)$ is a Gaussian variable and there are two sequences $x_t, y_t$ defined by the following update rules:*

- *$x_{t+1} \ge x_t + \eta C_t \mathbb{E}_\xi[(\xi + x_t)^p]$ for some $C_t = \Theta(1)$;*
- *$y_{t+1} \le y_t + \eta C_t \mathbb{E}_\xi[(\xi + y_t)^p]$ for some $S = \Theta(1)$.*

*Then if $x_0, y_0 \ge \mu, x_0 - y_0 \ge \varepsilon = \Omega(\frac{1}{\text{polylog}(d)})$, for every $t \ge 0$ such that $x_t \in [d^{0.01}x_0, O(1)]$, we shall have $y_t \le O(\text{polylog}(d))$.*

*Proof.* Let $\delta > 0$ be such that $\delta \gg \frac{\mu}{\sigma}$ and $\delta = o(1)$. Let $T_i, i \ge 0$ be defined as the first time when $x_t \ge (1 + \delta)^i x_0$, and $b := \min\{i | (1 + \delta)^i x_0 \ge A\}$. Now we can compute a growth lower bound for $\varepsilon_t := x_t - y_t$ at each step:

$$\varepsilon_{T_{i+1}} \ge \varepsilon_{T_i} + \sum_{t \in [T_i, T_{i+1})} \eta C_t \mathbb{E}_\xi[(\xi + x_t)^p - (\xi + y_t)^p]$$

$$\ge \varepsilon_{T_i} + \varepsilon_{T_i} \sum_{t \in [T_i, T_{i+1})} \eta C_t \mathbb{E}_\xi[(\xi + x_t)^{p-1}]$$

$$\ge \varepsilon_{T_i} + \varepsilon_{T_i} \sum_{t \in [T_i, T_{i+1})} \eta C_t \mathbb{E}_\xi[(\xi + x_t)^p] \cdot \frac{1}{(1 + \delta)x_{T_{i+1}}}$$

$$\ge \varepsilon_{T_i} + \varepsilon_{T_i} \delta(1 + \delta)^i x_0 \frac{1}{(1 + \delta)^{i+2}x_0}$$

$$\ge (1 + \frac{\delta}{(1 + \delta)^2})\varepsilon_{T_i}$$

Therefore let $i'$ be such that $(1 + \delta)^{i'} \ge \frac{\mu}{\varepsilon}$, we have that

$$\varepsilon_{T_{i'}} \ge (1 + \frac{\delta}{(1 + \delta)^2})^{i'} \varepsilon_0 \ge (1 - o(1))\mu \gg \sigma\sqrt{\log d} \tag{36}$$

At $t = T_{i'}$ we have $y_{T_{i'}} \le x_{T_{i'}} \le (1 + o(1))\frac{\mu}{\varepsilon}x_0$. After $t = T_{i'}$, we can construct a surrogate sequence $\widetilde{y}_t \ge y_t$ for $t \ge T_{i'}$ by initializing $\widetilde{y}_{T_{i'}} = y_{T_{i'}} + \frac{\mu}{2}$, and updates by

$$\widetilde{y}_{t+1} = \widetilde{y}_t + \eta C_t \widetilde{y}_t^p$$

This update guarantees that $\widetilde{y}_t \ge y_t$ because $\widetilde{y}_t^p \ge \mathbb{E}_\xi[(\xi + y_t)^p]$ at every step. Now by applying corollary E.1 to the sequence of $x_t, \widetilde{y}_t$ using the fact that $x_{T_{i'}} \ge \widetilde{y}_{T_{i'}} + \frac{\mu}{2} = \widetilde{y}_{T_{i'}}(1 + \frac{1}{\text{polylog}(d)})$, we can get the desired result for $\widetilde{y}_t$ and thus $y_t$ at $t = T_b$, i.e, when $x_t \ge A$. □

**Lemma E.13** (TPM, adapted from [81]). *Consider an increasing sequence $x_t \ge 0$ defined by $x_{t+1} = x_t + \eta C_t x_t^{q-1}$ for some integer $q \ge 3$ and $C_t = \Theta(1) > 0$, then we have for every $A > x_0$,*

*for every $\delta > 0$, and every $\eta \in (0, 1)$:*

$$\sum_{t \geq 0, x_t \leq A} \eta C_t \geq \left( \frac{\delta(1+\delta)^{-1}}{(1+\delta)^{q-2}-1} \left( 1 - \left( \frac{(1+\delta)x_0}{A} \right)^{q-2} \right) - \frac{O(\eta A^{q-1})}{x_0} \frac{\log(A/x_0)}{\log(1+\delta)} \right) \cdot \frac{1}{x_0^{q-2}}$$

$$\sum_{t \geq 0, x_t \leq A} \eta C_t \leq \left( \frac{(1+\delta)^{q-2}}{q-2} + \frac{O(\eta A^{q-1})}{x_0} \frac{\log(A/x_0)}{\log(1+\delta)} \right) \cdot \frac{1}{x_0^{q-2}}$$

This lemma has a corollary:

**Corollary E.1** (TPM, from [81]). *Let $q \geq 3$ be a constant and $x_0, y_0 = o(1)$ and $A = O(1)$. Let $\{x_t, y_t\}_{t \geq 0}$ be two positive sequences updated as*

- $x_{t+1} = x_t + \eta C_t x_t^{q-1}$ *for some $C_t = \Theta(1)$;*
- $y_{t+1} = y_t + \eta S C_t y_t^{q-1}$ *for some $S = \Theta(1)$.*

*Suppose $x_0 \geq y_0 S^{\frac{1}{q-2}}(1 + \frac{1}{\text{polylog}(d)})$, letting $T_x$ be the first iteration s.t., $x_t \geq A$, then $y_{T_x} \leq \widetilde{O}(y_0)$.*

# F   Learning Symmetry Group Actions

**Roadmap.** We begin by introducing notation and defining fibers, followed by a description of the learning curriculum and the key appearance events. We then break the training process into clearly defined phases and prove the main results for each phase: In Phase I, features emerge and compete; In Phase II, features grow and mutually cancel, with precise tracking of when different outcomes occur; In Phase III, features converge to a desired shape.

Let us restate the action structure of $\mathcal{G}$ and $\mathcal{Y}$ here.

**Assumption F.1** (Assumption 5.1, restated). Let **LEGO**$(\mathcal{X}, \mathcal{G}, \mathcal{Y})$ be the LEGO language, where $|\mathcal{Y}| = n_Y$[8] and $\mathcal{G}$ is a group acting on on $\mathcal{Y}$. We assume the (left) group action $\alpha : \mathcal{G} \times \mathcal{Y} \to \mathcal{Y}$ is transitive but not free. In particular, we assume $n_Y = |\mathcal{Y}| \in [\omega(1), O(\log d)]$ and the group $\mathcal{G}$ satisfy $n_G = |\mathcal{G}| = \Theta(\text{polylog}(d)) > \frac{1}{\varrho}$ for all $y \in \mathcal{Y}$.

**Remark F.1.** *When the action $\alpha : \mathcal{G} \times \mathcal{Y} \to \mathcal{Y}$ is transitive, there is only one orbit $\mathcal{Y} = \{g \cdot y \mid g \in \mathcal{G}\} \simeq \mathcal{G}/G_y$ for all $y \in \mathcal{Y}$. By orbit-stabilizer theorem, the stabilizer$G_y$ at any point $y$ has the same cardinality.*

**Remark F.2.** *The canonical action of a symmetry group $S_n$ on $\mathbb{Z}_n$ satisfy Assumption F.1 with the choice $n_y = \Theta(\frac{\log \log d}{\log \log \log d})$ that keeps $n_y! = O(\text{polylog}(d))$.*

## F.1   Preliminaries

We work with fiber-indexed features. We first define fibers and super-combinations $\varphi$, then we define neuron features by its index $\psi = (j, r, \varphi)$, and finally set the curriculum and appearance events in this notation.

**Definition F.1** (Fiber of values). Assuming $\mathcal{G}$ follows Assumption F.1. For each $j \in \tau(\mathcal{Y})$ and $y \in \mathcal{Y}$, define the **fiber** of $j$ at $y$ as:

$$\text{Fiber}_{j,y} := \{g \in \mathcal{G} \mid \tau(g \cdot y) = j\}$$

which collects all group elements that send $y$ to $y' = \tau^{-1}(j)$.

**Remark F.3.** *We denote $\text{Fiber}_{j,y}$ to be the set that transport $y$ to $y' = \tau^{-1}(j)$ because it is the pre-image of the predictor map $\phi_y(\cdot) : g \mapsto \tau(g \cdot y)$ over $y \in \mathcal{Y}$. Algebraically the fibers are the left-cosets $\{gG_y\}_{g \in \mathcal{G}}$, where $G_y := \{g \in \mathcal{G} \mid g \cdot y = y\}$ is the stablizer subgroup. Moreover, we have the following fact from the orbit-stablizer theorem.*

**Fact F.1** (cardinality of the fibers). For every $j \in \tau(\mathcal{Y})$ and $y \in \mathcal{Y}$, $|\text{Fiber}_{j,y}| = |\mathcal{G}|/|\mathcal{Y}| = n_G/n_Y$ from the orbit-stabilizer theorem.

**Definition F.2** (Super-combinations). Let $(j, y) \in \tau(\mathcal{Y}) \times \mathcal{Y}$, we define the **super-combinations**, which is a set of combinations:

$$\varphi_{j,y} := \text{Fiber}_{j,y} \times \{y\} \subseteq \mathcal{G} \times \mathcal{Y}, \quad \varphi_{j,y}^1 = \text{Fiber}_{j,y}, \quad \varphi_{j,y}^2 = \{y\}$$

---

[8]   We use $n_y$ or $n_Y$ interchangeably in the proof below.

Let $\Phi := \cup_{j \in \tau(\mathcal{Y}),\ y \in \mathcal{Y}} \varphi_{j,y}$ be the full set. For a fixed class $j \in \tau(\mathcal{Y})$, we denote the **correct** set to be $\Phi_j^\star := \cup_{y \in \mathcal{Y}} \varphi_{j,y}$, and the **incorrect** set $\Phi_j^\dagger := \cup_{j' \neq j, y \in \mathcal{Y}} \varphi_{j,y}$. A base pair $\phi = (g, y)$ belongs to $\varphi_{j,y}$ if and only if $g \in \mathrm{Fiber}_{j,y}$. To simplify the proof, we also define for each $\varphi_{j,y} \subset \Phi$, the confounding combinations

$$\Phi_\varphi^\dagger = \bigcup_{j' \neq j \ \mathbf{XOR} \ y' \neq y} \varphi_{j',y'}$$

That is, the set of combinations that intersect with exactly one component with $\varphi_{j,y}$.

**Definition F.3** (Neuron feature indices)**.** Let us index neuron features by $\psi = (j, r, \varphi)$ with $\varphi \in \Phi$ and $(j, r) \in \tau(\mathcal{Y}) \times [m]$, and define

$$\Psi_{j,r}^\star := \{(j, r, \varphi) : \varphi \in \Phi_j^\star\}, \quad \Psi_{j,r}^\dagger := \{(j, r, \varphi) : \varphi \in \Phi_j^\dagger\}.$$

And we further define

$$\Psi := \tau(\mathcal{Y}) \times [m] \times \Phi, \quad \Psi^\star = \bigcup_{(j,r) \in \tau(\mathcal{Y}) \times [m]} \Psi_{j,r}^\star, \quad \Psi^\dagger = \bigcup_{(j,r) \in \tau(\mathcal{Y}) \times [m]} \Psi_{j,r}^\dagger$$

The set $\Psi$ contains all the neuron indices we care about, and $\Psi^\star$ and $\Psi^\dagger$ contain the correct and incorrect indices for class $j$.

$V$**-Notations** We call a pair $\phi = (g, y)$ with $g \in \mathcal{G}$ and $y \in \mathcal{Y}$ a *base pair*, or following the naming of appendix E, a *base combination*. For neuron $(j, r)$, we define

$$V_{j,r}(g) := \langle \mathbf{W}_{5,j,r,2}, e_g \rangle, \quad V_{j,r}(y) := \langle \mathbf{W}_{5,j,r,5}, e_y \rangle, \quad V_{j,r}(\phi) := \tfrac{1}{2}\big(V_{j,r}(g) + V_{j,r}(y)\big).$$

For a fixed value $y$ and class $j$, the fiber $\mathrm{Fiber}_{j,y}$ collects all permutations sending $y$ to $\tau^{-1}(j)$. For $\psi = (j, r, \varphi_{j,y})$, we define:

$$V_\psi := \frac{1}{n_G/n_Y} \sum_{g \in \mathrm{Fiber}_{j,y}} V_{j,r}(g, y), \quad \overline{V}_\psi := \max_{g \in \mathrm{Fiber}_{j,y}} V_{j,r}(g, y), \quad \underline{V}_\psi := \min_{g \in \mathrm{Fiber}_{j,y}} V_{j,r}(g, y).$$

For compactness of notation, let us also write $V_{j,r}(x) = \sum_{p=3,4} \langle \mathbf{W}_{5,j,r,1}, e_{x_0} \rangle + \langle \mathbf{W}_{5,j,r,1}, e_{x_1} \rangle$ which is a random variable depending on the randomness of $x_0, x_1$.

The definition of learning curriculum for the symmetry group actions is essentially the same with the simply transitive case, with a slight difference of using the fiber-combination $\varphi$ instead of the base combination $\phi$. We repeat here for reference.

**Definition F.4** (Learning curriculum, symmetry group actions)**.** We generate an feature index set $\Sigma^\star$ over fiber-indexed features. Letting $\Sigma_0 = \Psi$, at each $i \in [n_y^2]$, we choose

$$\psi_i = \arg \max_{\psi = (j, r, \varphi) \in \Sigma_i,\ \varphi \in \Phi_j^\star} V_\psi^{(0)}.$$

Write $\psi_i = (j, r, \varphi)$, we define the next iteration set $\Sigma_{i+1}$ by excluding:

(1) The confusing combinations for $\varphi$ in the same neuron:

$$\Sigma_{\psi_i}^{\dagger,1} \equiv \Sigma_i^{\dagger,1} = \{(j, r, \varphi') \in \Sigma_i, \varphi' \in \Phi_\varphi^\dagger\};$$

(2) All other combinations in the same neuron $(j, r)$:

$$\Sigma_{\psi_i}^{\dagger,2} \equiv \Sigma_i^{\dagger,2} = \{(j, r, \varphi') \mid \varphi' \in \Phi \setminus (\Phi_\varphi^\dagger \cup \{\varphi\})\};$$

(3) The same combination $\varphi_{j,y}$ in other neurons of the same class $j, r', r' \neq r$:

$$\Sigma_{\psi_i}^{\dagger,3} \equiv \Sigma_i^{\dagger,3} = \{(j, r', \varphi_{j,y}) \in \Sigma_i : r' \neq r\}.$$

The next iteration is given by

$$\Sigma_{i+1} = \Sigma_i \setminus (\Sigma_{\psi_i}^{\dagger,1} \cup \Sigma_{\psi_i}^{\dagger,2} \cup \Sigma_{\psi_i}^{\dagger,3} \cup \{\psi_i\})$$

Eventually we obtain $\Sigma^\star = (\psi_1, \ldots, \psi_{n_y^2})$ and $\Sigma^\dagger := \bigcup_{\psi \in \Sigma^\star} \Sigma_\psi^\dagger$. We write $\psi \prec_\Sigma \psi'$ (or simply $\psi \prec \psi'$) if $\psi$ precedes $\psi'$ in $\Sigma^\star$. More generally, we also write $\psi \prec \psi'$ for $\psi, \psi' \in \Psi$ if $\psi \in \Sigma^\star$ and $\psi' \in \cup_{\psi'' \succ \psi} \Sigma_{\psi''}^\dagger$ is from the .

**Events of combination appearance.** These events specify whether a specific base combination $\phi = (g, y)$ appears, or only one of its component matches.

**Definition F.5** (Events of appearance). For $\phi = (g, y) \in \varphi_{j,y}$ for some $j \in \tau(\mathcal{Y})$ and $y \in \mathcal{Y}$, we define
$$\mathcal{H}_\phi := \{g_1 = g,\ y_0 = y\}, \quad \mathcal{H}_g := \{g_1 = g\}, \quad \mathcal{H}_y := \{y_0 = y\},$$

For the mismatched events $\mathcal{H}^\dagger$, the definition is a little different from the simply transitive actions: since $\varphi_{j,y} = \mathrm{Fiber}_{j,y} \times \{y\}$ and $g \in \mathrm{Fiber}_{j,y}$, we write
$$\mathcal{H}_\varphi^\dagger(g) := \{g_1 = g, y_0 \neq y\}, \quad \mathcal{H}_\varphi^\dagger(y) := \{g_1 \notin \mathrm{Fiber}_{j,y},\ y_0 = y\},$$

When considering the event of all the base pair $\phi \in \varphi_{j,y}$, we also define the union events
$$\mathcal{H}_\varphi := \bigcup_{\phi \in \varphi} \mathcal{H}_\phi, \quad \mathcal{H}_\varphi^\dagger := \mathcal{H}_\varphi^\dagger(y) \cup \left( \bigcup_{\phi = (g,y) \in \varphi} \mathcal{H}_\varphi^\dagger(g) \right).$$

### F.1.1 Theorem Statement

Now we present the theorem for learning symmetric group actions.

**Theorem F.1** (Learning Symmetric Group Actions). *Assuming Assumption 5.1, and let $F^{(t)}$ be obtained by Algorithm 2 at $t = T_1$, then the loss for the 5th-token is minimized, i.e., $\mathsf{Loss}_5^{(T_1)} \leq \frac{1}{\mathrm{poly}(d)}$, and for each $\psi = (j, r, \varphi) \in \Sigma^\star$, the followings hold:*

A. *For any $\phi \in \varphi$, we have $V_{j,r}^{(T_1)}(\phi) \in [B - O(d^{c_1}\mu), B + O(\mu)]$;*

B. *For any $\psi' = (j, r, \varphi') \in \Sigma^{\dagger,1}$ and $\phi' \in \varphi'$, we have $|V_{j,r}^{(T_1)}(\phi')| \leq \widetilde{O}(\varpi^{\frac{1}{q-1}})$. Or more concretely:*
$$\frac{1}{2}\left| V_{j,r}^{(T_1)}(g') + V_{j,r}^{(T_1)}(y') \right| \leq \widetilde{O}(\varpi^{\frac{1}{q-1}}), \quad \forall (g', y') \in \varphi'$$

C. *For any $\phi = (g, y) \in \varphi$ the following relation holds:*
$$\left| V_{j,r}^{(T_1)}(g) - C_\alpha V_{j,r}^{(T_1)}(y) \right| \leq \widetilde{O}(\varpi^{\frac{1}{q-1}})$$

*where $C_\alpha = \frac{1 + n_G(n_Y - 1)/n_Y}{(n_Y - 1) + n_G/n_Y} = \Theta(n_Y)$ due to our choice of $n_G, n_Y$ in Assumption F.1.*

D. *For any $\psi' \in \Sigma^{\dagger,3}$, it holds that $\overline{V}_{\psi'}^{(t)} = \widetilde{O}(\mu)$.*

The assertions of Theorem F.1 described the *feature shapes* of the neurons, which we formally defined in the proof of learning simply transitive actions (Definition E.1). The main difference between the results here and Theorem E.1 is that Theorem F.1C described a different symmetry of features without the factor $\Theta(n_y)$.

Let us follow the proof in Appendix E and the feature shape for symmetry group actions.

**Definition F.6** (Feature shape). Let $\psi = (j, r, \varphi) \in \Psi$, and let $\delta = (\delta_1, \delta_2)$ be the error parameters. We say the feature $\psi$ has shape $\mathcal{F}_\psi(\delta)$ if the followings are satisfied:

1. $\mathcal{F}_{\psi,1}(\delta_1)$: $\underline{V}_\psi^{(t)} \geq B - O(\delta_1)$ and $\overline{V}_\psi^{(t)} \leq B + O(\delta_1)$;

2. $\mathcal{F}_{\psi,2}(\delta_2)$: For any $\psi' = (j, r, \varphi') \in \Sigma_\psi^{\dagger,1}$, it holds that $|V_{j,r}^{(t)}(\phi')| \leq O(\delta_2)$ for all $\phi' \in \varphi'$;

3. $\mathcal{F}_{\psi,3}(\delta_2)$: $|V_{j,r}^{(t)}(g) - C_\mathcal{Y} V_{j,r}^{(t)}(y)| \leq O(\delta_2)$ for any $\phi = (g, y) \in \varphi$;

4. $\mathcal{F}_{\psi,4}$: For any feature $\psi' \in \Sigma_\psi^{\dagger,3}$, it holds that $\overline{V}_{\psi'}^{(t)} \leq \widetilde{O}(\mu)$.

Indeed, we aim to prove that $\mathcal{F}_\psi(\delta)$ holds for suitable parameter $\delta$ at the end of training, for every $\psi \in \Sigma^\star$. And this is indeed what the theorem states.

**Remark F.4.** *A major difference between Definition F.6 and Definition E.4 is that now the conditions involve not just one combination of weights, but all combinations $\phi \in \varphi$ given a combination $\varphi \in \Phi$. This creates some significant difficulties for the proof.*

### F.1.2 Facts and Basic Calculations

We now define the gradient notation and conditions.

**Definition F.7** (gradient notation). Let $\psi = (j, r, \varphi)$, we write $\Gamma^+$ and $\Gamma^-$ to denote the different terms in the gradient computation. For any base combination $\phi = (g, y) \in \varphi_{j,y}$ where $\varphi_{j,y} \in \Phi$, we write

$$\Gamma^{+,(t)}_{j,r,\phi} := \mathbb{E}\big[(1 - \mathbf{logit}_{5,j})\mathbf{sReLU}'(\Lambda_{5,j,r})\mathbb{1}_{\mathcal{H}_\phi}\big], \quad \Gamma^{-,(t)}_{j,r,\phi} := \mathbb{E}\big[\mathbf{logit}_{5,j}\mathbf{sReLU}'(\Lambda_{5,j,r})\mathbb{1}_{\mathcal{H}_\phi}\big].$$

When we consider the gradient of $V_{j,r}(g)$ or $V_{j,r}(y)$ for individual tokens $g \in \mathcal{G}$ and $y \in \mathcal{Y}$, we use the following notations: let $\phi = (g, y) \in \varphi_{j,y}$, then we define

$$\Gamma^{+,(t)}_{j,r,g} := \mathbb{E}\big[\mathbf{logit}_{5,j}\mathbf{sReLU}'(\Lambda_{5,j,r})\mathbb{1}_{\mathcal{H}_\phi}\big]$$

$$\Gamma^{-,(t)}_{j,r,g} := \mathbb{E}\big[\mathbf{logit}_{5,j}\mathbf{sReLU}'(\Lambda_{5,j,r})\mathbb{1}_{\mathcal{H}^\dagger_\varphi(g)}\big]$$

$$\Gamma^{-,(t)}_{j,r,y} := \mathbb{E}\big[\mathbf{logit}_{5,j}\mathbf{sReLU}'(\Lambda_{5,j,r})\mathbb{1}_{\mathcal{H}^\dagger_\varphi(y)}\big].$$

Finally, we sum up over $\phi \in \varphi$ for the gradient of combination $\psi = (j, r, \varphi)$:

$$\Gamma^{+,(t)}_\psi := \mathbb{E}\big[(1 - \mathbf{logit}_{5,j})\mathbf{sReLU}'(\Lambda_{5,j,r})\mathbb{1}_{\mathcal{H}_\varphi}\big],$$

$$\Gamma^{-,(t)}_\psi := \mathbb{E}\big[\mathbf{logit}_{5,j}\mathbf{sReLU}'(\Lambda_{5,j,r})\mathbb{1}_{\mathcal{H}^\dagger_\varphi}\big].$$

Let $\psi = (j, r, \varphi)$ where $\varphi_1 = \mathrm{Fiber}_{j,y}$ One can check that the defined $\Gamma^{+,(t)}_\psi$ is connected to the base pair $\Gamma$ by the following relation:

$$\Gamma^{+,(t)}_\psi = \sum_{g \in \mathrm{Fiber}_{j,y}} \Gamma^{+,(t)}_{j,r,g}, \quad \Gamma^{-,(t)}_\psi = \Gamma^{-,(t)}_{j,r,y} + \sum_{g \in \mathrm{Fiber}_{j,y}} \Gamma^{-,(t)}_{j,r,g} \tag{37}$$

**Definition F.8** (gradient condition). At $t \leq T_1$ and $\delta > 0$, write for $\psi = (j, r, \varphi)$

$$\mathcal{C}_\psi(\delta): \ |\Gamma^{+,(t)}_\psi - \Gamma^{-,(t)}_\psi| \leq \delta, \qquad \mathcal{C}^+_\psi(\delta): \ |\Gamma^{+,(t)}_\psi| \leq \delta, \qquad \mathcal{C}^-_\psi(\delta): \ |\Gamma^{-,(t)}_\psi| \leq \delta.$$

These conditions control the magnitude of the gradient of the feature at iteration $t$. Next we compute the expressions of the gradient using

**Fact F.2** (gradient expressions). Equipped with Definition F.8, the gradient with respect to $\mathbf{W}_{i,j,r,p}$ for $i = 5$, $j \in [d]$, $r \in [m]$, $p \in [5]$ and $v \in \mathcal{G} \cup \mathcal{Y}$ can be computed by

(a) when $j \in \tau(\mathcal{Y}), r \in [m], p = 5$, for $g \in \mathcal{G}$, let $\phi = (g, y) \in \varphi_{j,y}$ be the combination with $\phi_1 = g$, we have

$$\langle -\nabla_{\mathbf{W}_{5,j,r,2}}\mathsf{Loss}, e_g\rangle = \frac{1}{2}(\Gamma^{+,(t)}_{j,r,\phi} - \Gamma^{-,(t)}_{j,r,g}) \tag{38}$$

(b) when $j \in \tau(\mathcal{Y}), r \in [m], p = 5$, for $y \in \mathcal{Y}$, let $\varphi_{j,y} \in \Phi^\star_j$ be the combination, we have

$$\langle -\nabla_{\mathbf{W}_{5,j,r,5}}\mathsf{Loss}, e_y\rangle = \frac{1}{2}(\Gamma^{+,(t)}_\psi - \Gamma^{-,(t)}_{j,r,y}) \tag{39}$$

(c) For any combination of $p$, $v$ not included above, we have due to our assumptions of the distribution and update rule:

$$\langle -\nabla_{\mathbf{W}_{5,j,r,p}}\mathsf{Loss}, e_v\rangle \equiv 0$$

### F.1.3 Induction Hypothesis and Training Phases

We define the following intermediate time-steps:

**Definition F.9** (Phase decomposition). Define timestamps for $\psi$ using the feature:

(a) Phase I: $t \in [0, T^1_\psi]$, where $T^1_\psi := \min\{t \geq 0 : \underline{V}^{(t)}_\psi \geq d^{c_1}\mu\}$.

(b) Phase II.1: $t \in (T^1_\psi, T^{2,1}_\psi]$, where $T^{2,1}_\psi := \min\{t \geq 0 : V^{(t)}_\psi \geq \frac{1}{2}\log d\}$.

(c) Phase II.2: $t \in (T^{2,1}_\psi, T^{2,2}_\psi]$, where $T^{2,2}_\psi := \min\{t \geq 0 : \mathbb{E}[\mathcal{E}^{(t)}_{5,j}\mathbb{1}_{\mathcal{H}_\varphi}] \leq d^{-1/2}\}$.

(d) Phase II.3: $t \in (T_\psi^{2,2}, T_\psi^2]$, where $T_\psi^2 = \min\{T_\psi^{2,3}, T_{2,1} + O(\frac{1}{\eta(d^{c_1}\mu)^{q-2}})\}$, and $T_\psi^{2,3}$ is defined by

$$T_\psi^{2,3} := \min\{t \geq 0 : \mathcal{F}_\psi(\delta_1, \delta_2) \text{ holds, where } \delta_1 = d^{c_1}\mu, \delta_2 = \widetilde{O}((d\varpi/\lambda)^{\frac{1}{q-1}})\}$$

(e) Phase III.1: $t \in (T_\psi^2, T_{1,1}]$, where $T_{1,1} = T_{\psi_{n_y^2}}^2$ and $\psi_{n_y^2}$ is the last feature in $\Sigma^\star$.

(f) Phase III.2: $t \in (T_{1,1}, T_1]$, the end of the training where the feature shape has stabilized

We give several induction hypotheses, each characterizing different aspects of the process.

**Induction F.1** (learning symmetric group actions). *Let $t \leq T_1$. The following holds:*

(a) $\underline{V}_\psi^{(t)} + b_{i,j,r} \geq \Omega(\mu)$ *for all $\psi \in \Sigma^\star$.*

(b) $T_\psi^2 < T_{\psi'}^1$, *if $\psi \prec \psi' \in \Sigma^\star$. Thus the intervals $\{[T_\psi^1, T_\psi^2]\}_{\psi \in \Sigma^\star}$ are non-overlapping.*

(c) *Let $\psi' \prec \psi \in \Sigma^\star$, then the feature shape $\mathcal{F}_{\psi'}(d^{c_1}\sigma_0, \varpi^{\frac{1}{q-1}})$ holds throughout $t \in [T_\psi^1, T_1]$;*

(d) *For any $\psi = (j, r, \varphi) \in \Sigma^\star$ There are at most $\widetilde{O}(\sqrt{d}/\eta)$ iterations where $\mathbf{logit}^{(t)} \geq \frac{1}{\sqrt{d}}$ conditioned on $\mathcal{H}_\varphi$ in $t \in [0, T_\psi^1]$.*

**Interpretations of Induction F.1.** We explain what the inductions hypothesis mean here:

(a) This property simply asserts that the *good* features never shrink below a certain level which is crucial for the optimization to work.

(b) This property describes the *curriculum* nature of $\Sigma^\star$: each feature $V_\psi$ grows in different time intervals within $0 \leq t \leq T_1$.

(c) This property asserts that the feature $\psi'$ previous to $\psi$ in the curriculum $\Sigma^\star$ has stable and good feature shapes, which helps with learning the current feature.

(d) This property means there are very few iteration where the gradient of $V_{j,r}(\phi), \phi \in \varphi$ is disturbed before $T_{j,r,\varphi}^1$ is reached.

## F.2 Phase I: Emergence of the Feature

We first prove some properties at the beginning of phase I.

**Lemma F.1** (Initialization range for features). *Under random initialization and Assumption F.1, with probability at least $1 - o(1)$, the following holds*

(a) *For any $v \in \mathcal{V}$, we have $|\langle \mathbf{W}_{5,j,r,p}^{(0)}, e_x \rangle| \leq O(\mu/\sqrt{\log d})$ for any $j, r, p$.*

(b) *Let $\psi \in \Sigma^\star, \psi' \in \Psi$ and $\psi \prec \psi'$, then $V_\psi^{(0)} \geq V_{\psi'}^{(0)} + \Delta_0$ for a gap $\Delta_0 = \Omega(\sigma_0/\text{polylog}d)$.*

(c) *We have $\Lambda_{5,j,r}^{(0)}(\mathbf{Z}) = \Omega(\mu)$ for all $j \in [d], r \in [m]$.*

*Proof.* The proof is basically the same as that of Fact E.3. The first one uses basic concentration properties of Gaussian distribution and the second uses a union bound with the anti-concentration property of the Gaussian distribution. (c) is based on the fact that

$$\Lambda_{5,j,r}^{(0)}(\mathbf{Z}) = b_{5,j,r} + \frac{1}{2} \sum_{\mathbf{k} \in \mathcal{I}^{1,0}} \sum_{p \in [5]} \langle \mathbf{W}_{5,j,r,p}^{(0)}, \mathbf{Z}_{\mathbf{k},p} \rangle = \mu \pm O(\mu/\sqrt{\log d})$$

which concludes the proof. □

**Corollary F.1.** *For any $\psi = (j, r, \varphi) \in \Sigma^\star$ and $y \in \varphi^2$, we have with probability $\geq 1 - o(1)$:*

$$\left| \frac{1}{|\varphi|} \sum_{g \in \varphi^1} \mathbb{E}_x[\mathbf{sReLU}'(\Lambda_{5,j,r}^{(0)}) \mid \mathcal{H}_{g,y}] - \mathbb{E}_x[\mathbf{sReLU}'(V_{j,r}^{(0)}(y) + V_{j,r}(x) + b_{5,j,r})] \right|$$

$$\leq O(\frac{1}{\sqrt{n_G/n_Y}})\mathbb{E}_x[\mathbf{sReLU}'(V_{j,r}^{(0)}(y) + V_{j,r}(x) + b_{5,j,r})]$$

*Proof.* This is true because of lemma F.1a and Hoeffding's inequality. □

Now we present an induction for phase I.

**Induction F.2.** *Let $\psi = (j, r, \varphi) \in \Sigma^\star$, for $t \leq T_\psi^1$, the gradients satisfy $\Gamma_{j,r,y}^{+,(t)} \geq \Omega(\frac{n_G}{n_Y})\Gamma_{i,j,g}^{+,(t)}$ for any $(g, y) \in \varphi$.*

Using Fact E.1 with a representative $\phi = (g, y) \in \varphi_{j,y}$ and the definition $V_{j,r}(\phi) = \frac{1}{2}(V_{j,r}(g) + V_{j,r}(y))$, a single gradient step with step size $\eta$ yields

$$V_{j,r}^{(t+1)}(\phi) - V_{j,r}^{(t)}(\phi) = \frac{\eta}{2}\left(\langle -\nabla_{\mathbf{W}_{5,j,r,5}}\mathsf{Loss}, e_y\rangle + \langle -\nabla_{\mathbf{W}_{5,j,r,2}}\mathsf{Loss}, e_g\rangle\right)$$
$$= \frac{\eta}{2}\left(\Gamma_\psi^{+,(t)} + \Gamma_{j,r,\phi}^{+,(t)} - \Gamma_{j,r,g}^{-,(t)} - \Gamma_{j,r,y}^{-,(t)}\right) \tag{40}$$

Similar to the proof of the simply transitive actions, we define a proxy term and argue that the trajectory of $V_\psi$ is similar to the proxy.

**Lemma F.2** (Approximating the gradient with proxy). *Assuming Inductions F.1 and F.2 holds at $t$. Let $\psi = (j, r, \varphi) \in \Psi$, for every $\phi = (g, y) \in \varphi$, we define two proxy sequences $\widetilde{V}_{j,r}^{(t)}(v), v \in \{g, y\}$ by the update rule:*

$$\widetilde{V}_{j,r}^{(t+1)}(y) = \widetilde{V}_{j,r}^{(t)}(y) + \frac{\eta}{2}\widetilde{\Gamma}_\psi^{+,(t)}, \quad \widetilde{V}_{j,r}^{(t+1)}(g) = \widetilde{V}_{j,r}^{(t)}(g) + \frac{\eta}{2}\widetilde{\Gamma}_{j,r,g}^{+,(t)};$$

*where both $\widetilde{V}_{j,r}(g)$ and $\widetilde{V}_{j,r}(y)$ start from $\widetilde{V}_{j,r}^{(0)}(y) := V_{j,r}^{(0)}(y)$, $\widetilde{V}_{j,r}^{(0)}(g) := V_{j,r}^{(0)}(g)$. And the $\widetilde{\Gamma}$ are defined by replacing the features of $V_{j,r}(v), v \in \varphi^1 \cup \{y\}$ with $\widetilde{V}_{j,r}(v)$. Now for all $t \leq T_\psi^1$, we have*

$$|\widetilde{V}_{j,r}^{(t)}(\phi) - V_{j,r}^{(t)}(\phi)| \leq \widetilde{O}(\mu/d^{\frac{1}{2}-3c_1}), \quad \text{where } \widetilde{V}_{j,r}^{(t)}(\phi) = \widetilde{V}_{j,r}(g) + \widetilde{V}_{j,r}(y).$$

*Proof.* Fix $\psi = (j, r, \varphi_{j,y})$. (40) gives

$$|\widetilde{V}_{j,r}^{(t)}(y) - V_{j,r}^{(t)}(y)| = \frac{\eta}{2}\Gamma_{j,r,y}^{-,(t)}, \qquad |\widetilde{V}_{j,r}^{(t)}(g) - V_{j,r}^{(t)}(g)| = \frac{\eta}{2}\Gamma_{j,r,g}^{-,(t)}$$

Since Induction F.2 holds at $t$, we have that $\widetilde{V}_{j,r}^{(t+1)}(y) - \widetilde{V}_{j,r}^{(t)}(y) \geq \Omega(\frac{n_G}{n_Y})(\widetilde{V}_{j,r}^{(t+1)}(g) - \widetilde{V}_{j,r}^{(t)}(g))$. Moreover, since $\sum_{g \in \varphi^1}\Gamma_{j,r,g}^{+,(t)} = \Gamma_{j,r,y}^{+,(t)}$ by Definition F.7. This, combined with Induction F.1, implies that

$$\widetilde{V}_{j,r}^{(t+1)}(y) - \widetilde{V}_{j,r}^{(t)}(y) \geq \sum_{g \in \varphi^1}\Gamma_{j,r,g}^{+,(t)} \geq \Omega(\frac{n_G}{n_Y})(\widetilde{V}_{j,r}^{(t+1)}(g) - \widetilde{V}_{j,r}^{(t)}(g)), \quad \forall g \in \varphi^1$$

The above update bound imply, for all $t \leq T_\psi^1$, the following property hold: $\widetilde{V}_{j,r}^{(t)}(g) - \widetilde{V}_{j,r}^{(0)}(g) \leq O(\frac{1}{n_G/n_Y})(\widetilde{V}_{j,r}^{(t)}(y) - \widetilde{V}_{j,r}^{(0)}(y))$ for all $g \in \varphi^1$. Now, by the same induction procedure in the proof of Lemma E.2, we know that the sum

$$\left|V_{j,r}^{(t)}(y) + \sum_{g \in \varphi^1} V_{j,r}^{(t)}(g) - \widetilde{V}_{j,r}^{(t)}(y) + \sum_{g \in \varphi^1} \widetilde{V}_{j,r}^{(t)}(g)\right| \leq O(\mu/d^{\frac{1}{2}-3c_1})$$

And then by the above relations we know

$$|\widetilde{V}_{j,r}^{(t)}(y) - V_{j,r}^{(t)}(y)| \leq O(\mu/d^{\frac{1}{2}-3c_1})$$

Thus the conclusion holds. $\qquad\square$

**Lemma F.3** (competition). *Let $\psi \in \Sigma^\star$ Assuming Inductions F.1 and F.2 holds at $t \leq T_\psi^1$. Then at $t = T_\psi^1$, we have for any feature $\psi' \in \Psi$, $\psi' \succ \psi$, $\overline{V}_{\psi'}^{(t)} \leq \widetilde{O}(\mu)$.*

*Proof.* From lemma F.2, we know that

$$\widetilde{V}_{j,r}^{(t+1)}(y) = \widetilde{V}_{j,r}^{(t)}(y) \geq \frac{\eta}{2} \sum_{g \in \varphi^1} \mathbb{E}_x[(\widetilde{V}_{j,r}^{(t)}(y) + \widetilde{V}_{j,r}^{(t)}(g) + b_{5,j,r} + V_{j,r}(x))^{q-1}\mathbb{1}_{\mathcal{H}_{g,y}}]$$

Suppose there is another $\psi' = (\varphi', y') \in \Sigma^\star$ such that $\psi' \succ \psi$, then we can also write a comparison based on lemma F.1:

$$\widetilde{V}_{j,r}^{(0)}(y) \geq \widetilde{V}_{j,r}^{(0)}(y') + \Delta_0,$$

And therefore one can compute by absolutely bounding the average over $g \in \varphi$ using corollary F.1:

$$\Gamma_\psi^{+,(0)} = \frac{1}{n_Y |\varphi^1|} \sum_{g \in \varphi^1} \mathbb{E}_x[(\widetilde{V}_{j,r}^{(0)}(y) + \widetilde{V}_{j,r}^{(0)}(g) + b_{5,j,r} + V_{j,r}(x))^{q-1}]$$

$$\geq \frac{1}{n_Y}(1 - \frac{1}{\sqrt{n_G/n_Y}})\mathbb{E}_x[(\widetilde{V}_{j,r}^{(0)}(y) + b_{5,j,r} + V_{j,r}(x))^{q-1}]$$

Similarly, for $\psi' = (j', r', \varphi')$ and $\phi' \in \varphi'$, we have

$$\Gamma_{\psi'}^{+,(0)} = \frac{1}{|\varphi'^1|} \sum_{g \in \varphi'^1} \mathbb{E}_x[(\widetilde{V}_{j',r'}^{(0)}(y') + \widetilde{V}_{j',r'}^{(0)}(g) + b_{5,j',r'} + V_{j',r'}(x))^{q-1} \mid \mathcal{H}_{g,y'}]$$

$$\leq (1 + \frac{1}{\sqrt{n_G/n_Y}})\mathbb{E}_x[(\widetilde{V}_{j',r'}^{(0)}(y') + b_{5,j',r'} + V_{j',r'}(x))^{q-1}]$$

Now we can compare the proxies $\widetilde{V}_{j,r}(y) + b_{5,j,r}$ with $\widetilde{V}_{j,r}(y')$ using lemma E.12 to get that when $V_{j,r}(y) \geq d^{c_1}\mu$ we have $V_{j',r'}(y') \leq \widetilde{O}(\mu)$. Now due to lemma F.2 and Induction F.2, we have the desired result. $\qquad\square$

*Proof of Inductions F.1 and F.2 in Phase I.* Fix $\psi = (j, r, \varphi) \in \Sigma^\star$. We check items (a)-(d) in Induction F.1:

- Induction F.1a: This is simple by combining lemmas F.1 and F.2, where each $V_{j,r}(\phi)$ is well approximated by a monotonically growing proxy $\widetilde{V}_{j,r}(\phi)$.
- Induction F.1b: This is proved later but here lemma F.3 showed that for any pair $\psi \prec \psi' \in \Sigma^\star$, that $T_\psi^1 < T_{\psi'}^1$, which is a precondition for Induction F.1b.
- Induction F.1c-d: These two are proved later in future phases and are not violated in phase I.
- Induction F.2: We shall show that this is the case. Consider any feature $g \in \varphi^1$, its gradient is clearly bounded by $O(n_Y/n_G)\Gamma_\psi^{+,(t)}$ initially before the feature $V_{j,r}(g) \geq \Omega(\mu)$. In this range the difference of its growth with the update of $V_{j,r}(y)$ averaged over all $g \in \varphi^1$ is bounded by $O(n_Y/n_G)(\widetilde{V}_{j,r}^{(t)}(y) + b_{5,j,r})^{q-1}$ because it grows slower than $O(n_Y/n_G)\widetilde{V}_{j,r}(y)$ and thus Hoeffding's inequality with variance parameter $\mu$ would work to bound the difference between the averaged gradient and the gradient with $V_{j,r}(g)$ in expectation, and thus we obtain a bound $\Gamma_\psi^{+,(t)} \geq \Omega(n_G/n_Y)\Gamma_{j,r,g}^{+,(t)}$. When $V_{j,r}(g) \geq \Omega(\mu)$, the feature $V_{j,r}(y)$ has grown more than $\Omega(n_G/n_Y)\mu$ and the growth ratio still holds. This proved the induction for $t \leq T_\psi^1$.

$\qquad\square$

### F.3 Phase II: Feature Growth and Cancellations

Below we present an additional induction hypothesis for phase II.

**Induction F.3** (Phase II). *For all features $\psi = (j, r, \varphi) \in \Sigma^\star$ and $t \in [T_\psi^1, T_\psi^2]$, then for any $g' \notin \varphi^1, y' \notin \varphi^2$, we have $V_{j,r}^{(t)}(g'), V_{j,r}^{(t)}(y') \in (-B - \widetilde{O}(\mu), \widetilde{O}(\mu))$.*

#### F.3.1 Technical Lemmas

**Lemma F.4** (gradient estimation). *Assume Inductions F.1 and F.3. Let $\psi = (j, r, \varphi_{j,y}) \in \Sigma^\star$, let $\phi = (g, y) \in \varphi_{j,y}$, then at $t \in [T_\psi^1, T_\psi^2]$, we have*

*(a) For $t \in [T_\psi^1, T_\psi^{2,1}]$, we have $\Gamma_\psi^{(t)} \geq \Omega(\mathbf{Pr}(\mathcal{H}_\varphi))\mathbf{sReLU}'(\underline{V}_\psi^{(t)})$;*

*(b) If $V_{j,r}^{(t)}(\phi) \in (\frac{1}{3}\log d, \frac{2}{3}\log d)$, then $\Gamma_{j,r,\phi}^{(t)} = \Gamma_{j,r,\phi}^{+,(t)} - \Gamma_{j,r,\phi}^{-,(t)} \geq \Omega(\mathbf{Pr}(\mathcal{H}_\phi))$;*

(c) For any $g' \notin \varphi^1$ and any $y' \neq y$,

$$\Gamma_{j,r,g'}^{(t)} = -\mathbb{E}\big[\mathbf{logit}_{5,j}^{(t)}\mathbf{sReLU}'(\Lambda_{5,j,r}^{(t)})\mathbb{1}_{\mathcal{H}_{(g',y)}}\big] \pm O(\varpi n_Y/n_G),$$

$$\Gamma_{j,r,y'}^{(t)} = -\sum_{g\in\varphi^1}\mathbb{E}\big[\mathbf{logit}_{5,j}^{(t)}\mathbf{sReLU}'(\Lambda_{5,j,r}^{(t)})\mathbb{1}_{\mathcal{H}_{(g,y')}}\big] \pm O(\varpi).$$

(d) Any individual component $V_{j,r}^{(t)}(v), v \in \varphi^1 \cup \varphi^2$ cannot fall below $O(\mu)$, and $\Gamma_{j,r,v}^{+,(t)} \geq 0$.

*Proof.* Write $\psi = (j, r, \varphi_{j,y})$.

(a) By the definition of $T_\psi^{2,1}$, we know there exist $\phi \in \varphi$ such that $V_{j,r}(\phi) \leq \frac{1}{2}\log d$. By the same arguments in lemma F.2, we can get that all $\phi \in \varphi$ satisfy $\frac{1}{2}\log d + o(1)$. Given that Induction F.1 holds at $t$, conditioned on $\mathcal{H}_\phi$, all other neuron $(j', r') \neq (j, r)$ has activation smaller than $\varrho$. Therefore $F_{5,j'} \leq o(1)$, and $\mathbf{logit}_{5,j'} = o(1)$ for any $j' \neq j$ conditioned on $\mathcal{H}_\phi$. So conditioned on $\mathbf{Z} \in \mathcal{H}_\phi$, when $V_{j,r}^{(t)}(\phi) \leq \frac{2}{3}\log d$, we have

$$\mathbf{logit}_{5,j} = \frac{e^{F_{5,j}(\mathbf{Z})}}{e^{F_{5,j}(\mathbf{Z})} + (1 + o(1))(d - 1)} \leq O(d^{-\Omega(1)})$$

Now we can simply bound all $\Gamma_{j,r,v}^{-,(t)}$ for $v \in \varphi^1 \cup \varphi^2$ to be smaller than $\widetilde{O}(\frac{1}{\sqrt{d}})\Gamma_{j,r,v}^{+,(t)}$. Thus the proof is obtained by computing $\Gamma_{j,r,v}^{+,(t)}$ and sum to get $\Gamma_\psi^{+,(t)}$.

(b) Similar to (a), here we also notice that $\mathbf{sReLU}'(\Lambda_{5,j,r}(\mathbf{Z})) = 1$ because $\Lambda_{5,j,r}(\mathbf{Z}) \in [\varrho, B]$ is in the linear regime. Now we can see that the terms $\Gamma_{j,r,\phi}^{+,(t)} \geq \Omega(\mathbf{Pr}(\mathcal{H}_\phi))$ and $\Gamma_{j,r,\phi}^{-,(t)} \leq O(d^{-\Omega(1)})$ and thus we have the result.

(c) For $g' \notin \mathrm{Fiber}_{j,y}$, let $y' \in \mathcal{Y}$ such that $g' \in \mathrm{Fiber}_{j,y'}$, then we have

$$\Gamma_{j,r,g'}^{(t)} = \mathbb{E}[(1 - \mathbf{logit}_{5,j})\mathbf{sReLU}'(\Lambda_{5,j})\mathbb{1}_{\mathcal{H}_{(g',y')}}] - \sum_{y' \neq y', y}\mathbb{E}[\mathbf{logit}_{5,j}\mathbf{sReLU}'(\Lambda_{5,j})\mathbb{1}_{\mathcal{H}_{(g',y')}}]$$

$$- \mathbb{E}[\mathbf{logit}_{5,j}\mathbf{sReLU}'(\Lambda_{5,j})\mathbb{1}_{\mathcal{H}_{(g',y)}}]$$

Since $\mathcal{H}_{(g',y)}$ is confusing, the negative term dominates; other appearances contribute at most $O(\varpi n_Y/n_G)$ or $O(\mu^{q-1})$. Similar arguments hold for $y'$.

(d) This one is proven similarly to (a) but can adapt to any level of $V_{j,r}(\phi)$ for the corresponding $\phi \in \varphi$ for $v \in \varphi^1 \cup \varphi^2$. From Induction F.3c we know that all other feature $V_{j,r}(v') \leq \widetilde{O}(\mu)$ for $v' \notin \varphi^1 \cup \varphi^2$. So for any $v \in \varphi^1 \cup \varphi^2$ such that $V_{j,r}(v) \leq \widetilde{O}(\mu)$, we have that the combination $\phi = (v, v') \in \Phi \setminus \varphi$ has $V_{j,r}(\phi) < O(\mu)$ and thus have gradient

$$\Gamma_{j,r,v}^{(t)} = \Gamma_{j,r,v}^{+,(t)} - \Gamma_{j,r,v}^{-,(t)} \geq \Gamma_{j,r,v}^{+,(t)} - O(\mu^{q-1})$$

Note that $\Gamma_{j,r,v}^{+,(t)}$ is either $\geq \lambda/d$ or $0$ at this phase. So we can sum up the negative gradient $-O(\mu^{q-1})$ for all iterations $t \in [T_\psi^1, T_\psi^2]$ (which is fewer than $\widetilde{O}(\frac{1}{\eta(d^{c_1}\mu)^{q-2}})$) to get that the feature $V_{j,r}(v) \gg \mu$. This means $\Lambda_{5,j,r} \geq 0$ when $\mathcal{H}_\phi$ happens and thus $\Gamma_{j,r,v}^{+,(t)} \geq 0$ for all $t \in [T_\psi^1, T_\psi^2]$.

$\square$

**Lemma F.5** (feature magnitude). *Assume Inductions F.1 and F.3 holds. For any feature $\psi = (j, r, \varphi) \in \Sigma^\star$, we have*

(a) *when $\mathcal{C}_\psi^+(\delta)$ holds for some $\delta \in [n_G\varpi, O(\lambda/d^{c_1})]$ at $t \geq T_\psi^{2,1}$, we have $V_{j,r}^{(t)}(\phi) \geq B - O(d^{c_1}\mu), \forall \phi \in \varphi$.*

(b) *when $\mathcal{C}_\psi^-(\delta)$ holds for some $\delta \in [n_G\varpi, O(\lambda/d^{1+c_1})]$ at $t \geq T_\psi^{2,1}$, we have $V_{j,r}^{(t)}(\phi) \in [-B - O(\mu), \widetilde{O}((\frac{d\delta}{\lambda})^{\frac{1}{q-1}})]$ for all $\phi \in \Phi_\varphi^\dagger$.*

*Proof.*      (a) If $\mathcal{C}_\psi^+(\delta)$ holds with $\delta \leq O(\lambda/d^{c_1})$, then there are two probability: $V_{j,r}^{(t)}(\phi)$ for all $\phi \in \varphi$ is as small as $O(\delta^{\frac{1}{q-1}})$ which is not possible for $\delta$ so small because of lemma F.4a; or that $V_{j,r}^{(t)}(\phi) \geq B - O(d^{c_1}\mu)$ such that $F_{5,j} \geq B$ and thus the gradient has vanished for some events.

(b) The logic from (a) can be applied here as well. Since the positive gradient is at most $O(\varpi)$ by lemma F.4b, thus whenever $\delta \gg \varpi$, the negative gradient conditioned on $\mathcal{H}_\phi, \phi \in \Phi_\varphi^\dagger$ is bounded, which gives

$$\sum_{\phi \in \varphi} \mathbb{E}[\textbf{logit}_{5,j}\textbf{sReLU}'(\Lambda_{5,j,r})\mathbb{1}_{\mathcal{H}_\phi}] \leq O(\delta) \implies (\Lambda_{5,j,r})^{q-1}\mathbb{1}_{\mathcal{H}_\phi} \leq \delta\lambda/(d\mathbf{Pr}(\mathcal{H}_\phi))$$

(because Fact E.2)

$$\implies V_{j,r}^{(t)}(\phi) \lesssim \left(\frac{n_G n_Y \lambda\delta}{d}\right)^{\frac{1}{q-1}}, \forall \phi \in \varphi$$

which proves the desired result.

$\square$

**Lemma F.6** (gradient stationarity). *Let* $\psi = (j, r, \varphi) \in \Sigma^\star$, *then for any* $\delta > \varpi^{q-2}$, *the followings hold:*

(a) *Define* $\mathcal{B}_\delta^+ := \{t \in [T_\psi^{2,1}, T_\psi^{2,3}] \mid \mathcal{C}_\psi^+(\delta) \text{ doesn't hold}\}$ *then* $|\mathcal{B}_\delta^+| \leq O(\frac{\log^2 d}{\eta\delta})$;

(b) *Define* $\mathcal{B}_\delta^+ := \{t \in [T_\psi^{2,1}, T_\psi^{2,3}] \mid \mathcal{C}_\psi^-(\delta) \text{ doesn't hold}\}$ *then* $|B_\delta^-| \leq O(\frac{\log^3 d}{\eta\delta})$;

(c) *For some* $t = T_\psi^{2,3} \geq T_\psi^{2,1} + O(\frac{1}{\eta n_G\varpi})$, *we have* $V_{j,r}^{(t)}(\phi) \geq B - O(d^{c_1}\mu)$ *for all* $\phi \in \varphi$ *and* $|V_{j,r}^{(t)}(\phi')| \leq \widetilde{O}((\frac{d\varpi}{\lambda})^{\frac{1}{q-1}})$ *for all* $\phi' \in \Phi_\varphi^\dagger$, *moreover, this will hold until* $t = T_\psi^2$

*Proof.* The proof is very similar in spirit to the proof of Lemma E.6. We first define the following quantity:

$$\Upsilon_\psi^{(t)} = V_{j,r}^{(t)}(y) - \sum_{g \notin \varphi^1} V_{j,r}^{(t)}(g)$$

One can compute that the update of $\Upsilon_\psi^{(t)}$ is simply

$$\eta\left(\Gamma_\psi^{+,(t)} - \sum_{\varphi' \neq \varphi \in \Phi_j^\star} \sum_{\phi \in \varphi'} \Gamma_{j,r,\phi}^{+,(t)} + \sum_{\phi' \notin \varphi \cup \Phi_\varphi^\dagger} \Gamma_{j,r,\phi'}^{-,(t)}\right)$$

The middle term is bounded by $O(\varpi)$ and the last term is bounded below by $-O(\varpi)$, thus we have that the update satisfy

$$\Upsilon_\psi^{(t+1)} - \Upsilon_\psi^{(t)} \geq \eta\Gamma_\psi^{+,(t)} - O(\eta\varpi)$$

Since $\Gamma_\psi^{+,(t)} \geq 0$ For any $\delta \gg \varpi$ by lemma F.4c, we know that the above bound accumulate at $t \geq T_\psi^{2,1}$ at

$$\Upsilon_\psi^{(t)} - \Upsilon_\psi^{(T_\psi^{2,1})} \geq \sum_{s \in [T_\psi^{2,1}, t] \cup B_\delta^+} \eta\delta - O(\eta\varpi)$$

So for no more than $O(\frac{\log^2 d}{\eta\delta})$ iterations we should have the that $\Upsilon^{(t)} \geq \Omega(\log^2 d)$, which is impossible in our setting. This contradiction proves the result for (a). The proof of (b) is similar to (a) and the proof of lemma E.6b. (c) is also similar to lemma E.6c by using lemma F.5a-b with parameter $\lambda/d^{c_1}$ and parameter $n_G\varpi$, alonge with lemma F.7. We do not repeat the details here. $\square$

**Lemma F.7** (Symmetry of features). *Let* $\psi = (j, r, \varphi) \in \Sigma^\star$ *and* $t \geq T_\psi^{2,3}$, *we have that if* $\mathcal{F}_{\psi,2}(\delta)$ *hold for some* $\delta > 0$, *then*

$$|V_{j,r}^{(t)}(g) - C_\alpha V_{j,r}^{(t)}(y)| \leq O(\delta)$$

*For some* $C_\alpha = \frac{1+n_G(n_Y-1)/n_Y}{(n_Y-1)+n_G/n_Y}$.

*Proof.* For a feature $\psi = (j, r, \varphi) \in \Sigma^\star$, their update can be represented as:

$$V_{j,r}^{(t)}(g) - V_{j,r}^{(0)}(g) = U_{g,y} - \sum_{y' \neq y} R_{g,y'}$$

$$V_{j,r}^{(t)}(y) - V_{j,r}^{(0)}(y) = \sum_{g \in \mathrm{Fiber}_{j,y}} U_{g,y} - \sum_{g' \notin \mathrm{Fiber}_{j,y}} R_{g',y}$$

where $U_{g,y} = \sum_{s \leq t} \eta \Gamma_{j,r,\phi}^{+,(s)}$ and $R_{g,y'}$ is

$$R_{g,y'} = \sum_{s \leq t} \eta \mathbb{E}[\mathbf{logit}_{5,j}^{(s)} \mathbf{sReLU}'(\Lambda_{5,j,r}^{(s)}) \mathbb{1}_{\mathcal{H}_{g',y}}]$$

Therefore:

$$\sum_{g \in \mathrm{Fiber}_{j,y}} V_{j,r}^{(t)}(g) = \sum_{g \in \mathrm{Fiber}_{j,y}} U_{g,y} - \sum_{g \in \mathrm{Fiber}_{j,y}} \sum_{y' \neq y} R_{g,y'}$$

Thus it is true for any $\mathrm{Fiber}_{j,y}$ and $y \in \mathcal{Y}$ that the following holds

$$V_{j,r}^{(t)}(y) - V_{j,r}^{(0)}(y) + \sum_{g' \notin \mathrm{Fiber}_{j,y}} R_{g',y} = \sum_{g \in \mathrm{Fiber}_{j,y}} (V_{j,r}^{(t)}(g) - V_{j,r}^{(0)}(g)) + \sum_{g \in \mathrm{Fiber}_{j,y}} \sum_{y' \neq y} R_{g,y'}$$

By summing both the left and right hand side with all the $(\mathrm{Fiber}_{j,y}, y)$ pairs, we have

$$\sum_{y \in \mathcal{Y}} (V_{j,r}^{(t)}(y) - V_{j,r}^{(0)}(y)) + \sum_{y \in \mathcal{Y}} \sum_{g' \notin \mathrm{Fiber}_{j,y}} R_{g',y} = \sum_{g \in \mathcal{G}} (V_{j,r}^{(t)}(g) - V_{j,r}^{(0)}(g)) + \sum_{g \in \mathcal{G}} \sum_{y' : \tau(g \cdot y') \neq j} R_{g,y'}$$

Since $\sum_{y \in \mathcal{Y}} \sum_{g' \notin \mathrm{Fiber}_{j,y}} R_{g',y} = \sum_{g \in \mathcal{G}} \sum_{y' : \tau(g \cdot y') \neq j} R_{g,y'}$, we have

$$\sum_{y \in \mathcal{Y}} (V_{j,r}^{(t)}(y) - V_{j,r}^{(0)}(y)) = \sum_{g \in \mathcal{G}} (V_{j,r}^{(t)}(g) - V_{j,r}^{(0)}(g))$$

$$\implies \sum_{y \in \mathcal{Y}} V_{j,r}^{(t)}(y) = \sum_{g \in \mathcal{G}} V_{j,r}^{(t)}(g) \pm O(\mu)$$

Moreover, we have assumed

$$|V_{j,r}^{(t)}(g) + V_{j,r}^{(t)}(y')| = O(\delta), \quad \forall y' \neq y, g \in \mathrm{Fiber}_{j,y},$$
$$|V_{j,r}^{(t)}(y) + V_{j,r}^{(t)}(g')| = O(\delta), \quad \forall g' \notin \mathrm{Fiber}_{j,y}$$

Thus

$$(1 + \frac{|\mathcal{G}|}{|\mathcal{Y}|}(|\mathcal{Y}| - 1))V_{j,r}^{(t)}(y) = (\frac{|\mathcal{Y}|(|\mathcal{Y}| - 1)}{|\mathcal{G}|} + 1) \sum_{g \in \mathrm{Fiber}_{j,y}} V_{j,r}^{(t)}(g) \pm O(n_G \delta)$$

Therefore choosing whatever $g \in \mathrm{Fiber}_{j,t}$, we have

$$(1 + \frac{|\mathcal{G}|}{|\mathcal{Y}|}(|\mathcal{Y}| - 1))V_{j,r}^{(t)}(y) = (\frac{|\mathcal{Y}|^2}{|\mathcal{G}|} + 1)\frac{|\mathcal{G}|}{|\mathcal{Y}|} V_{j,r}^{(t)}(g) \pm O(n_G \delta_2) = (|\mathcal{Y}| - 1 + \frac{|\mathcal{G}|}{|\mathcal{Y}|})V_{j,r}^{(t)}(g) \pm O(n_G \delta)$$

Now by dividing the right-hand side by the factor on $V_{j,r}^{(t)}(y)$ on the LHS, we have the desired result. $\qquad \square$

### F.3.2 Proof of Induction

**Lemma F.8** (Arrival times). *Assuming Inductions F.1 and F.3 for $\psi = (j, r, \varphi) \in \Sigma^\star$, we have*

(a) $T_\psi^{2,1} - T_\psi^1 \leq \widetilde{O}(\frac{1}{\eta(d^{c_1}\sigma_0)^{q-2}})$;

(b) $T_\psi^{2,2} - T_\psi^{2,1} \leq O(\frac{d^{\frac{1}{2}+c_1}}{\eta})$;

*(c)* $T_\psi^{2,3} \le \widetilde{O}(\frac{1}{\eta\varpi})$;

*(d)* $T_\psi^2 \le \widetilde{O}(\frac{1}{\eta(d^{2c_1}\mu)^{q-2}})$.

*Proof.*     (a) By lemma F.4a, after phase I, within $\widetilde{O}(1/(\eta(d^{c_1}\sigma_0)^{q-2}))$ we have $V_{j,r}^{(t)}(\phi) \ge \frac{1}{2}\log d$ for all $\phi \in \varphi$.

(b) On $[T^1, T^{2,1}]$, lemma F.6a gives the rate $O(d^{\frac{1}{2}+c_1}/\eta)$ time for the positive grad to arrive at $1 - \mathbf{logit}_{5,j} \le \frac{1}{\sqrt{d}}$ conditioned on $\mathcal{H}_\varphi$.

(c) This is again provided by lemma F.6c by using $\delta = n_G\varpi$, combined with lemma F.5b for the feature shape guarantees.

(d) This is simply by the definition of $T_\psi^2$ and that $T_\psi^{2,3} \ll \frac{1}{\eta(d^{2c_1}\mu)^{q-2}}$.

$\square$

*Proof of Inductions F.1 and F.3.* Fix $\psi = (j, r, \varphi_{j,y})$ and consider $t \in [T_\psi^1, T_\psi^2]$, then

- Induction F.1a: This is guaranteed by lemma F.4a-b.
- Induction F.1b: Because the time used is small between $T_\psi^1$ and $T_\psi^2$ by lemma F.8, we have that $T_{\psi'}^1$ is still behind when this happens.
- Induction F.1c: This will be proven in phase III, we do not use any properties that violate this hypothesis.
- Induction F.1d: This is due to Induction F.3a-b and lemma F.6a with parameter $\delta = 1/\sqrt{d}$.
- Induction F.3: For any $\phi \notin \varphi$, we have that they are either desirable features $\phi \in \varphi_{j,y'}$ for some different $y' \ge y \in \mathcal{Y}$, or that they are the wrong features for the class $j$, i.e., $\phi \in \varphi_{j',y}$ for all $y \in \mathcal{Y}$. The feature $\psi' = (j, r, \varphi_{j,y'})$ in the former case was out-competed by $\psi = (j, r, \phi)$ so their growth is bounded by $\widetilde{O}(\mu)$ in the first two stage and negative in the end due to the feature shape $\mathcal{F}_\psi(d^{c_1}\mu, (\frac{d\varpi}{\lambda})^{\frac{1}{q-1}})$ at the end as shown in lemma F.6c. Combination $\phi$ in the latter case will be smaller as well due to lemma F.6c.

The time bounds follow from lemma F.8. This proves the induction in phase II for any feature $\psi \in \Sigma^\star$.

$\square$

## F.4 Phase III: Convergence

We present the induction hypothesis in this phase.

**Induction F.4** (Phase III, final). *Let $\psi = (j, r, \varphi) \in \Sigma^\star$, for $t \in [T_\psi^2, T_1]$, the following holds:*

*(a) At $t \in [T_\psi^2, T_{1,1}]$, we have $\mathcal{F}_\psi(\delta_1, \delta_2)$ holds with $\delta_1 = d^{c_1}\mu$, $\delta_2 = \widetilde{O}((\frac{d\varpi}{\lambda})^{\frac{1}{q-1}})$;*

*(b) At $t \in [T_{1,1}, T_1]$, we have $\mathcal{F}_\psi(\delta_1, \delta_2)$ holds with $\delta_1 = d^{c_1}\mu$, $\delta_2 = \widetilde{O}((\varpi)^{\frac{1}{q-1}})$.*

We obtain the shape of logits at convergence.

**Lemma F.9** (Logit shape at convergence). *Assuming Induction F.4. Let $\psi_{n_y^2} =\in \Sigma^\star$ be the last of $\Sigma^\star$, then at $t \in [T_{\psi_{n_y^2}}^{2,3}, T_1]$, the followings hold:*

*(a) For $\phi \in \Phi_j^\star$, $\mathbf{logit}_{5,j}^{(t)} \ge 1 - O(\lambda)$ conditioned on $\mathcal{H}_\phi$.*

*(b) For $\phi \notin \Phi_j^\star$, $\mathbf{logit}_{5,j}^{(t)} \le O(\lambda/d)$ conditioned on $\mathcal{H}_\phi$.*

*Proof.* The proof is very similar to that of lemma E.9, we do not repeat here.     $\square$

**Lemma F.10** (Gradient bounds, phase III). *Assuming Inductions F.1 and F.4 holds, and let $\psi = (j, r, \varphi) \in \Sigma^\star$, then at $t \in [T_{1,1}, T_1]$, the followings hold*

*(a) $\Gamma_\psi^{(t)} \ge \Omega(\lambda)$ when $V_\psi^{(t)} \le B - d^{c_1}\mu$;*

*(b) $|\Gamma_{j,r,v}^{-,(t)}| \le O(\frac{\lambda\varpi}{dn_y})$ for any $v \in \mathcal{G} \cup \mathcal{Y}$;*

*(c) For $\phi \in \cup_{y \in \mathcal{Y}} \varphi_{j,y}$, we have*

$$\Gamma_{j,r,\phi}^{+,(t)} = \begin{cases} -\Theta(\lambda\varpi/n_y^2), & \text{if } V_{j,r,\phi'}^{(t)} \in [-B + 2\mu, -\varpi] \\ \in [-\Theta(\lambda\varpi/n_y^2), -\Theta(\lambda\varpi/dn_y^2)] \cup \{0\}, & \text{if } V_{j,r,\phi'}^{(t)} \in [-B - 2\mu, -B + 2\mu] \end{cases}$$

*Proof.* The proof of (a) is simple as $\lambda \gg \varpi$ which is the upper bound of all other gradient terms. (b) is because of lemma F.9b and that Induction F.4b holds which guaranteed suitable feature shape. (c) is simply a combination of (b) and the lemma F.9a, and Induction F.4b. □

### F.4.1 Proof of Induction in Phase III and theorem F.1

*Proof of Induction F.4 and theorem F.1.* The proof is similar to the proof of Induction E.4, thus we leave out some details here.

- Induction F.4a: We know from Induction F.3 and lemmas F.6 and F.8 that Induction F.4a holds at $t = T_\psi^2$. The feature shape $\mathcal{F}_\psi(d^{c_1}\mu, \widetilde{O}((\frac{d\varpi}{\lambda})^{\frac{1}{q-1}}))$ guaranteed a starting point for reusing the argument in lemma F.6c to guarantee that the feature stays at the same feature shape.

- Induction F.4b: After $t = T_{1,1}$, which is $t = T_{\psi_{n_y^2}}^2$ for the last $\psi \in \Sigma^\star$. We shall prove that the feature shape $\mathcal{F}_\psi(d^{c_1}\mu, \widetilde{O}((\varpi)^{\frac{1}{q-1}}))$ holds for every $\psi \in \Sigma^\star$. Since (a) holds for all $\psi \in \Sigma^\star$, we have that at $T_{\psi_{n_y^2}}^{2,3}$ that the feature shape in (a) holds for all $\psi = (j, r, \varphi) \in \Sigma^\star$. So by lemma F.10b and similar argument in lemma E.6c, we can prove that the feature $|V_{j,r}^{(t)}(\phi)| \leq \widetilde{O}(\varpi^{\frac{1}{q-1}})$ for any $\phi \in \Phi_\varphi^\dagger$, which is $\mathcal{F}_{\psi,2}(\widetilde{O}(\varpi^{\frac{1}{q-1}}))$. $\mathcal{F}_{\psi,1}(d^{c_1}\mu)$ is guaranteed by lemma F.10a and $\mathcal{F}_{\psi,3}(\widetilde{O}(\varpi^{\frac{1}{q-1}}))$ is guaranteed by $\mathcal{F}_{\psi,1}, \mathcal{F}_{\psi,2}$ and lemma F.7. $\mathcal{F}_{\psi,4}$ is simple as any feature $\psi \in \Sigma_\psi^{\dagger,3}$ lost the competition and are bounded by $\widetilde{O}(\mu)$. Moreover, once each $\varphi_{j,y}, j \in \tau(\mathcal{Y}), y \in \mathcal{Y}$ is learned. the feature in $\Sigma_\psi^{\dagger,3}$ has too small gradient bounded by $O(\frac{\lambda}{d}\mu^{q-1})$ which will not have sufficient growth before $t = T_1$.

Since all feature $\psi \in \Sigma^\star$ have shape $\mathcal{F}_\psi(\delta_1, \delta_2)$ for $\delta_1 = d^{c_1}\mu$ and $\delta_2 = \widetilde{O}(\varpi^{\frac{1}{q-1}})$ at $t = T_1$, we have proven theorem F.1. □

## G    Learning the Attention Layer: Simply Transitive Case

In this section, we consider the case where the group operations form a simply transitive group. According to Assumption 4.1, we assume that for any $y_1, y_2 \in \mathcal{Y}$, there exists a unique $g \in \mathcal{G}$ such that $g \cdot y_1 = y_2$. Without loss of generality, we let $\mathcal{Y} = \{0, 1, \ldots, n_y - 1\}$, where $n_y \in [\Omega(\log\log d), \log d]$.

We focus on updating only $\mathbf{Q}$, while keeping $\mathbf{W}$ fixed. Combined with the attention structure specified in Assumption C.2, it suffices to consider the updates to the blocks $\mathbf{Q}_{4,3}$ and $\mathbf{Q}_{4,4}$ only. We consider the contribution to the gradient from the position $i = 5$ on task $\mathcal{T}^2$; specifically, the relevant loss function is given by $\sum_{\ell=1}^2 \text{Loss}_5^{2,\ell}$. As $\mathbf{W}$ remains fixed in this section, we omit the superscript $(t)$ in $\mathbf{W}$ and in all related notations that depend solely on $\mathbf{W}$ (e.g., $V_{j,r}$) for notational simplicity.

### G.1    Gradient Computations

**Notations for gradient expressions.** We firsr introduce some notations for the gradients of the attention layer. For $1 \leq \ell \leq L$, given $\mathbf{Z}^{L,\ell-1}$ and $\mathbf{k} \in \mathcal{I}^{L,\ell-1}$, define

$$\Xi_{\ell,i,\mathbf{k}}^L(\mathbf{Z}^{L,\ell-1}) \triangleq \sum_{j \in [d]} \mathcal{E}_{i,j}(\mathbf{Z}^{L,\ell-1}) \sum_{r \in [m]} \mathbf{sReLU}'\big(\Lambda_{i,j,r}(\mathbf{Z}^{L,\ell-1})\big)\langle \mathbf{W}_{i,j,r}, \mathbf{Z}_\mathbf{k}\rangle, \quad i \in [5]. \quad (41)$$

For simplicity of notation, we will henceforth omit the dependence on $\mathbf{Z}^{L,\ell-1}$ in the notation of $\Xi_{\ell,i,\mathbf{k}}^L$ when it is clear from the context.

**Fact G.1** (Gradients of $\mathbf{Q}$). For any $p, q \in [5]$, we have

$$-\nabla_{\mathbf{Q}_{p,q}}\text{Loss}^L = \sum_{\ell=1}^L \sum_{i \in [5]} -\nabla_{\mathbf{Q}_{p,q}}\text{Loss}_i^{L,\ell}, \quad \text{where}$$

$$-\nabla_{\mathbf{Q}_{p,q}}\mathsf{Loss}_i^{L,\ell} =$$

$$\mathbb{E}\left[\sum_{\mathbf{k}\in\mathcal{I}^{L,\ell-1}}\mathbf{Attn}_{\mathsf{ans},\ell-1\to\mathbf{k}}\cdot\left(\Xi_{\ell,i,\mathbf{k}}^L - \sum_{\mathbf{k}'\in\mathcal{I}^{L,\ell-1}}\mathbf{Attn}_{\mathsf{ans},\ell-1\to\mathbf{k}'}\Xi_{\ell,i,\mathbf{k}'}^L\right)\mathbf{Z}_{\mathsf{ans},\ell-1,p}\mathbf{Z}_{\mathbf{k},q}^\top\right].$$

**Lemma G.1** (Gradients of $\mathbf{Q}_{4,3}$). *Given $s\in\tau(\mathcal{X})$, for the diagonal entry $[\mathbf{Q}_{4,3}]_{s,s}$ of the block $\mathbf{Q}_{4,3}$, we have*

$$\left[-\nabla_{\mathbf{Q}_{4,3}}\mathsf{Loss}_5^{2,1}\right]_{s,s} = \mathbb{E}\left[\mathbf{Attn}_{\mathsf{ans},0\to\mathsf{pred},1}\cdot\left(\sum_{j\in[d]}\mathcal{E}_{5,j}(\mathbf{Z}^{2,0})\sum_{r\in[m]}\mathbf{sReLU}'(\Lambda_{5,j,r})\cdot\right.\right.$$
$$\left.\left.\left(\langle\mathbf{W}_{5,j,r},\mathbf{Z}_{\mathsf{pred},1}\rangle - \Lambda_{5,j,r} + b_{5,j,r}\right)\right)\mathbb{1}_{s=\tau(x_0)}\right],$$

$$\left[-\nabla_{\mathbf{Q}_{4,3}}\mathsf{Loss}_5^{2,2}\right]_{s,s} = \mathbb{E}\left[\mathbf{Attn}_{\mathsf{ans},1\to\mathsf{pred},2}\cdot\left(\sum_{j\in[d]}\mathcal{E}_{5,j}(\mathbf{Z}^{2,1})\sum_{r\in[m]}\mathbf{sReLU}'(\Lambda_{5,j,r})\cdot\right.\right.$$
$$\left.\left.\left(\langle\mathbf{W}_{5,j,r},\mathbf{Z}_{\mathsf{pred},2}\rangle - \Lambda_{5,j,r} + b_{5,j,r}\right)\right)\mathbb{1}_{s=\tau(x_1)}\right].$$

*Moreover, for the off-diagonal entries $[\mathbf{Q}_{4,3}]_{s,s'}$ with $s\neq s'$, we have*

$$\left[-\nabla_{\mathbf{Q}_{4,3}}\mathsf{Loss}_5^{2,1}\right]_{s,s'} = \mathbb{E}\left[\mathbf{Attn}_{\mathsf{ans},0\to\mathsf{pred},2}\cdot\left(\sum_{j\in[d]}\mathcal{E}_{5,j}(\mathbf{Z}^{2,0})\sum_{r\in[m]}\mathbf{sReLU}'(\Lambda_{5,j,r})\cdot\right.\right.$$
$$\left.\left.\left(\langle\mathbf{W}_{5,j,r},\mathbf{Z}_{\mathsf{pred},2}\rangle - \Lambda_{5,j,r} + b_{5,j,r}\right)\right)\mathbb{1}_{s=\tau(x_0),s'=\tau(x_1)}\right],$$

$$\left[-\nabla_{\mathbf{Q}_{4,3}}\mathsf{Loss}_5^{2,2}\right]_{s,s'} = \mathbb{E}\left[\mathbf{Attn}_{\mathsf{ans},1\to\mathsf{pred},1}\cdot\left(\sum_{j\in[d]}\mathcal{E}_{5,j}(\mathbf{Z}^{2,1})\sum_{r\in[m]}\mathbf{sReLU}'(\Lambda_{5,j,r})\cdot\right.\right.$$
$$\left.\left.\left(\langle\mathbf{W}_{5,j,r},\mathbf{Z}_{\mathsf{pred},1}\rangle - \Lambda_{5,j,r} + b_{5,j,r}\right)\right)\mathbb{1}_{s=\tau(x_1),s'=\tau(x_0)}\right].$$

*Proof.* For $\ell = 1$, due to Fact G.1, the diagonal entry $[\mathbf{Q}_{4,3}]_{s,s}$ with $s\in\tau(\mathcal{X})$, the expected gradient contribution from $-\nabla_{\mathbf{Q}_{4,3}}\mathsf{Loss}_5^{2,1}$ takes the form

$$\mathbb{E}\left[\mathbf{Attn}_{\mathsf{ans},0\to\mathsf{pred},1}\cdot\left(\Xi_{\ell,5,\mathsf{pred},1}^2 - \sum_{\mathbf{k}'\in\mathcal{I}^{2,\ell-1}}\mathbf{Attn}_{\mathsf{ans},\ell-1\to\mathbf{k}'}\Xi_{\ell,5,\mathbf{k}'}^2\right)\cdot\mathbb{1}_{s=\tau(x_0)}\right],$$

which is nonzero in expectation only when $s = \tau(x_0)$. Therefore, combined the definition of $\Xi$ in (41) we have:

$$\left[-\nabla_{\mathbf{Q}_{4,3}}\mathsf{Loss}_5^{2,1}\right]_{s,s} = \mathbb{E}\left[\mathbf{Attn}_{\mathsf{ans},0\to\mathsf{pred},1}\cdot\left(\Xi_{\ell,5,\mathsf{pred},1}^2 - \sum_{\mathbf{k}'\in\mathcal{I}^{2,\ell-1}}\mathbf{Attn}_{\mathsf{ans},\ell-1\to\mathbf{k}'}\Xi_{\ell,5,\mathbf{k}'}^2\right)\mathbb{1}_{s=\tau(x_0)}\right]$$

$$= \mathbb{E}\left[\mathbf{Attn}_{\mathsf{ans},0\to\mathsf{pred},1}\cdot\left(\sum_{j\in[d]}\mathcal{E}_{5,j}\sum_{r\in[m]}\mathbf{sReLU}'(\Lambda_{5,j,r})\cdot\right.\right.$$
$$\left.\left.\left(\langle\mathbf{W}_{5,j,r},\mathbf{Z}_{\mathsf{pred},1}\rangle - \sum_{\mathbf{k}'\in\mathcal{I}^{2,\ell-1}}\mathbf{Attn}_{\mathsf{ans},\ell-1\to\mathbf{k}'}\langle\mathbf{W}_{5,j,r},\mathbf{Z}_{\mathbf{k}'}\rangle\right)\right)\mathbb{1}_{s=\tau(x_0)}\right]$$

$$= \mathbb{E}\left[\mathbf{Attn}_{\mathsf{ans},0\to\mathsf{pred},1}\cdot\left(\sum_{j\in[d]}\mathcal{E}_{5,j}\sum_{r\in[m]}\mathbf{sReLU}'(\Lambda_{5,j,r})\cdot\right.\right.$$

$$\left( \langle \mathbf{W}_{5,j,r}, \mathbf{Z}_{\mathsf{pred},1} \rangle - \Lambda_{5,j,r} + b_{5,j,r} \right) \right) \mathbb{1}_{s=\tau(x_0)} \Bigg].$$

Other quantities are computed similarly, and thus is omitted here. $\qquad\square$

**Lemma G.2** (Gradients of $\mathbf{Q}_{4,4}$). *Given $s \in \tau(\mathcal{X})$, for the diagonal entry $[\mathbf{Q}_{4,4}]_{s,s}$ of the block $\mathbf{Q}_{4,4}$, we have*

$$\left[ -\nabla_{\mathbf{Q}_{4,4}} \mathsf{Loss}_5^{2,1} \right]_{s,s} = \mathbb{E}\Bigg[ \mathbf{Attn}_{\mathsf{ans},0 \to \mathsf{ans},0} \cdot \Bigg( \sum_{j \in [d]} \mathcal{E}_{5,j}(\mathbf{Z}^{2,0}) \sum_{r \in [m]} \mathbf{sReLU}'(\Lambda_{5,j,r}) \cdot$$

$$\left( \langle \mathbf{W}_{5,j,r}, \mathbf{Z}_{\mathsf{ans},0} \rangle - \Lambda_{5,j,r} + b_{5,j,r} \right) \right) \mathbb{1}_{s=\tau(x_0)} \Bigg],$$

$$\left[ -\nabla_{\mathbf{Q}_{4,4}} \mathsf{Loss}_5^{2,2} \right]_{s,s} = \mathbb{E}\Bigg[ \mathbf{Attn}_{\mathsf{ans},1 \to \mathsf{ans},1} \cdot \Bigg( \sum_{j \in [d]} \mathcal{E}_{5,j}(\mathbf{Z}^{2,1}) \sum_{r \in [m]} \mathbf{sReLU}'(\Lambda_{5,j,r}) \cdot$$

$$\left( \langle \mathbf{W}_{5,j,r}, \mathbf{Z}_{\mathsf{ans},1} \rangle - \Lambda_{5,j,r} + b_{5,j,r} \right) \right) \mathbb{1}_{s=\tau(x_1)} \Bigg].$$

*Moreover, for the off-diagonal entries $[\mathbf{Q}_{4,4}]_{s,s'}$ with $s \neq s'$, we have $[-\nabla_{\mathbf{Q}_{4,4}} \mathsf{Loss}_5^{2,1}]_{s,s'} = 0$, and*

$$\left[ -\nabla_{\mathbf{Q}_{4,4}} \mathsf{Loss}_5^{2,2} \right]_{s,s'} = \mathbb{E}\Bigg[ \mathbf{Attn}_{\mathsf{ans},1 \to \mathsf{ans},0} \cdot \Bigg( \sum_{j \in [d]} \mathcal{E}_{5,j}(\mathbf{Z}^{2,1}) \sum_{r \in [m]} \mathbf{sReLU}'(\Lambda_{5,j,r}) \cdot$$

$$\left( \langle \mathbf{W}_{5,j,r}, \mathbf{Z}_{\mathsf{ans},0} \rangle - \Lambda_{5,j,r} + b_{5,j,r} \right) \right) \mathbb{1}_{s=\tau(x_1), s'=\tau(x_0)} \Bigg].$$

*Proof.* The analysis is similar to that of Lemma G.1, and we omit the details here. $\qquad\square$

### G.2 Some Useful Bounds for Gradients

In this subsection, we establish several useful bounds on the gradients of the attention layer, leveraging the feature structure of the MLP layer learned during stage 1. These bounds will be instrumental for the subsequent analysis.

As established in Lemma E.7 and Theorem E.1, at the end of stage 1, the network exhibits the following activation properties:

- **Sparse activations:** For each $j \in \tau(\mathcal{Y})$, and any feature $(g,y) \in \mathfrak{F}_j$, there exists a unique *activated* neuron $r \in [m]$ such that, under the event $g_1 = g$ and $y_0 = y$, the following holds:

$$\Lambda_{5,j,r}^{(T_1)} \geq B - O(\delta), \quad |V_{j,r}^{(T_1)}(g) - V_{j,r}^{(T_1)}(y)| \leq O(\delta)$$
$$\Lambda_{5,j,r'}^{(T_1)} \leq O(\delta^{q-1}) \quad \text{for all } r' \neq r.$$

- **Cancellation of incorrect features:** let $r \in [m]$ be the activated neuron associated with $(g,y) \in \mathfrak{F}_j$. Then for any $g' \neq g \in \mathcal{G}$ and any $y' \in \mathcal{Y}$, we have:

$$\left| V_{j,r}^{(T_1)}(g) + V_{j,r}^{(T_1)}(y') \right| \leq O(\delta),$$
$$\left| V_{j,r}^{(T_1)}(g') + V_{j,r}^{(T_1)}(y) \right| \leq O(\delta),$$

In the analysis of FFN layer, we have a pair $\delta = (\delta_1, \delta_2)$; in the expressions above, some terms should use $\delta_1$ and others $\delta_2$. For brevity, we write $\delta := \max\{\delta_1, \delta_2\}$ and bound all such terms by $\delta$. Since both $\delta_1$ and $\delta_2$ are small, this simplification does not affect our analysis.

**Notations for activated neurons.** Since in the simply transitive case, the feature sets are disjoint across indices—i.e., $\mathfrak{F}_j \cap \mathfrak{F}_{j'} = \emptyset$ for all $j \neq j'$—we denote the activated neuron corresponding to

$(g, y) \in \mathfrak{F}_j$ by $r_{g \cdot y}$. Moreover, let

$$\mathfrak{A} \triangleq \cup_{j \in \tau(\mathcal{Y})} \mathfrak{A}_j, \quad \text{where } \mathfrak{A}_j \triangleq \{r \mid \exists (g, y) \in \mathfrak{F}_j, r = r_{g \cdot y}\}.$$

In other words, $\mathfrak{A}$ is the set of all activated neurons across all feature sets $\mathfrak{F}_j$ for $j \in \tau(\mathcal{Y})$. Given $\mathbf{Z}^{L,\ell-1}$, letting $\widehat{\mathcal{G}}(\mathbf{Z}^{L,\ell-1}) = \cup \{g_{\ell'}\}_{\ell'=1}^{L}$ be the collection of all the chosen group elements in the predicate clauses. Similarly $\widehat{\mathcal{Y}} = \cup \{y_{\ell'}\}_{\ell'=0}^{\ell-1}$. Then define $\widehat{\mathfrak{A}}_j(\mathbf{Z}^{L,\ell-1}) = \{r_{g \cdot y} \mid (g, y) \in \mathfrak{F}_j \wedge (g \in \widehat{\mathcal{G}}(\mathbf{Z}^{L,\ell-1}) \vee y \in \widehat{\mathcal{Y}})\}$. For simplicity, we omit the dependence on $\mathbf{Z}^{L,\ell-1}$ in the notation of $\widehat{\mathfrak{A}}_j$ when it is clear from the context. Equipped with these notations, we can summarize the above properties in the following lemmas.

**Lemma G.3** (Properties of target feature magnitude). *Given $(g, y) \in \mathfrak{F}_j$ with $j \in \tau(\mathcal{Y})$, then, the following properties hold.*

$$\frac{1}{2} \left( V_{j,r_{g \cdot y}}(g) + V_{j,r_{g \cdot y}}(y) \right) \geq B - O(\delta), \quad \left| V_{j,r_{g \cdot y}}(g) - V_{j,r_{g \cdot y}}(y) \right| \leq O(\delta), \tag{42}$$

$$\left| V_{j,r_{g \cdot y}}(g) + V_{j,r_{g \cdot y}}(y') \right| \leq O(\delta), V_{j,r_{g \cdot y}}(y') < 0 \quad \text{for all } y' \neq y, \tag{43}$$

$$\left| V_{j,r_{g \cdot y}}(g') + V_{j,r_{g \cdot y}}(y) \right| \leq O(\delta), V_{j,r_{g \cdot y}}(g') < 0 \quad \text{for all } g' \neq g. \tag{44}$$

$$|V_{j,r}(g)|, |V_{j,r}(y)| \leq O(\delta) \quad \text{for all } r \notin \mathfrak{A}_j. \tag{45}$$

**Lemma G.4** (Properties of irrelevant magnitude). *If $(p, v) \notin \{2\} \times \mathcal{G} \cup \{5\} \times \mathcal{Y}$, or $j \notin \tau(\mathcal{Y})$, then for any $r \in [m]$, we have*

$$\left| \langle \mathbf{W}_{5,j,r,p}, e_v \rangle \right| \leq \widetilde{O}(\sigma_0). \tag{46}$$

The above lemmas give us some direct computations of the inner products between the weight matrices and input embedding vectors.

**Lemma G.5.** *Let $j \in \tau(\mathcal{Y})$ and $\ell \in [2]$. Then for any $r \in [m]$, the following holds:*

$$\langle \mathbf{W}_{5,j,r}, \mathbf{Z}_{\mathsf{pred},\ell} \rangle = V_{j,r}(g_\ell) \pm \widetilde{O}(\sigma_0), \tag{47}$$

$$\langle \mathbf{W}_{5,j,r}, \mathbf{Z}_{\mathsf{ans},\ell-1} \rangle = V_{j,r}(y_{\ell-1}) \pm \widetilde{O}(\sigma_0). \tag{48}$$

*Moreover, for $j \notin \tau(\mathcal{Y})$ and any $\mathbf{k} \in \mathcal{I}^{2,1}$ and $r \in [m]$, we have*

$$\left| \langle \mathbf{W}_{5,j,r}, \mathbf{Z_k} \rangle \right| = \widetilde{O}(\sigma_0). \tag{49}$$

*Proof.* By direct computations and the definition of $\mathbf{Z}_{\mathsf{pred},\ell}$, we have

$$\langle \mathbf{W}_{5,j,r}, \mathbf{Z}_{\mathsf{pred},\ell} \rangle = \langle \mathbf{W}_{5,j,r,1}, e_{x_\ell} \rangle + \langle \mathbf{W}_{5,j,r,2}, e_{g_\ell} \rangle + \langle \mathbf{W}_{5,j,r,3}, e_{x_{\ell-1}} \rangle$$

Plug in Lemma G.4 and the definition of $V_{j,r}(g_\ell)$, we obtain (47). The proof of (48) and (49) is similar, and we omit the details here. $\square$

Furthermore, we can establish some characterizations of the $\Lambda_{5,j,r}(\mathbf{Z}^{2,\ell-1})$ quantities, which are crucial for the following analysis.

**Lemma G.6** (Characterizations of Lambda). *Given $\mathbf{Z}^{2,\ell-1}$ with $\ell \in [2]$, with an attention structure $\{\mathbf{Attn}_{\mathsf{ans},\ell-1 \to \mathbf{k}}\}_{\mathbf{k} \in \mathcal{I}^{2,\ell-1}}$,*

*(a) for $j \in \tau(\mathcal{Y})$, for activated neuron $r \in \mathfrak{A}_j$, we have*

$$\Lambda_{5,j,r}(\mathbf{Z}^{2,\ell-1}) = \sum_{\ell'=1}^{2} \mathbf{Attn}_{\mathsf{ans},\ell-1 \to \mathsf{pred},\ell'} V_{j,r}(g_{\ell'}) + \sum_{\ell'=1}^{\ell} \mathbf{Attn}_{\mathsf{ans},\ell-1 \to \mathsf{ans},\ell'-1} V_{j,r}(y_{\ell'-1}) \pm \widetilde{O}(\sigma_0).$$

*(b) for $j \in \tau(\mathcal{Y})$, for any non-activated neuron $r \notin \mathfrak{A}_j$ we have*

$$\left| \Lambda_{5,j,r}(\mathbf{Z}^{2,\ell-1}) \right| \leq O(\delta).$$

*(c) for $j \notin \tau(\mathcal{Y})$, for any $r \in [m]$, we have*

$$\left| \Lambda_{5,j,r}(\mathbf{Z}^{2,\ell-1}) \right| \leq \widetilde{O}(\sigma_0).$$

*Proof.* Recall the definition of $\Lambda_{5,j,r}(\mathbf{Z}^{2,\ell-1})$ in (23), we have

$$\Lambda_{5,j,r}(\mathbf{Z}^{2,\ell-1}) = \sum_{\mathbf{k}\in\mathcal{I}^{2,\ell-1}} \mathbf{Attn}_{\mathsf{ans},\ell-1\to\mathbf{k}} \cdot \langle \mathbf{W}_{5,j,r}, \mathbf{Z_k}\rangle + b_{5,j,r}.$$

Thus, the first part follows directly from (47) and (48); similarly, the second part holds by plugging (45) into (47) and (48); the last part is a direct consequence of (49) and the fact $b_{i,j,r} = \sigma_0 \log d$.

$\square$

A direct consequence of the above lemma is the following finer characterization of the activated neurons.

**Lemma G.7.** *Given $j \in \tau(\mathcal{Y})$, for $r \in \mathfrak{A}_j \setminus \widehat{\mathfrak{A}}_j$, we have $\mathbf{sReLU}'(\Lambda_{5,j,r}) = 0$.*

*Proof.* For $j \in \tau(\mathcal{Y})$, for $r \in \mathfrak{A}_j$, by Lemma G.6, we have

$$\Lambda_{5,j,r}(\mathbf{Z}^{2,\ell-1}) = \sum_{\ell'=1}^{2} \mathbf{Attn}_{\mathsf{ans},\ell-1\to\mathsf{pred},\ell'} V_{j,r}(g_{\ell'}) + \sum_{\ell'=1}^{\ell} \mathbf{Attn}_{\mathsf{ans},\ell-1\to\mathsf{ans},\ell'-1} V_{j,r}(y_{\ell'-1}) \pm \widetilde{O}(\sigma_0).$$

By (45), for for $r \in \mathfrak{A}_j \setminus \widehat{\mathfrak{A}}_j$, we have $V_{j,r}(g_{\ell'}), V_{j,r}(y_{\ell'-1}) \leq -B - O(\delta)$. Hence

$$\Lambda_{5,j,r} = -\Omega(B) \pm \widetilde{O}(\sigma_0) \ll -\varrho,$$

which implies $\mathbf{sReLU}'(\Lambda_{5,j,r}) = 0$.

$\square$

Now we are ready to further derive the gradients of the attention layer starting from Lemmas G.1 and G.2 and the properties established above.

**Lemma G.8** (Refined expression for the gradient of $\mathbf{Q}_{4,3}$). *Given $s \in \tau(\mathcal{X})$, for the diagonal entry $[\mathbf{Q}_{4,3}]_{s,s}$ of the block $\mathbf{Q}_{4,3}$, letting $j_1 = \tau(g_1(y_0))$ and $j_2 = \tau(g_2(y_1))$, we have*

$$\left[-\nabla_{\mathbf{Q}_{4,3}}\mathsf{Loss}_5^{2,1}\right]_{s,s} = \mathbb{E}\Bigg[\mathbf{Attn}_{\mathsf{ans},0\to\mathsf{pred},1}\cdot$$

$$\left((1-\mathbf{logit}_{5,j_1})\cdot\Big(\sum_{r\in\widehat{\mathfrak{A}}_{j_1}}\mathbf{sReLU}'(\Lambda_{5,j_1,r})\cdot\Big(V_{j_1,r}(g_1)-\Lambda_{5,j_1,r}\pm\widetilde{O}(\sigma_0)\Big)\pm\widetilde{O}(\delta^q)\Big)\right.$$

$$-\sum_{j\neq j_1\in\tau(\mathcal{Y})}\mathbf{logit}_{5,j}\cdot\Big(\sum_{r\in\widehat{\mathfrak{A}}_j}\mathbf{sReLU}'(\Lambda_{5,j,r})\cdot\Big(V_{j,r}(g_1)-\Lambda_{5,j,r}\pm\widetilde{O}(\sigma_0)\Big)\pm\widetilde{O}(\delta^q)\Big)$$

$$\left.\pm\sum_{j\notin\tau(\mathcal{Y})}\mathbf{logit}_{5,j}\widetilde{O}(\sigma_0^q)\right)\mathbb{1}_{\tau(x_0)=s}\Bigg];$$

$$\left[-\nabla_{\mathbf{Q}_{4,3}}\mathsf{Loss}_5^{2,2}\right]_{s,s} = \mathbb{E}\Bigg[\mathbf{Attn}_{\mathsf{ans},1\to\mathsf{pred},2}\cdot$$

$$\left((1-\mathbf{logit}_{5,j_2})\cdot\Big(\sum_{r\in\widehat{\mathfrak{A}}_{j_2}}\mathbf{sReLU}'(\Lambda_{5,j_2,r})\cdot\Big(V_{j_2,r}(g_2)-\Lambda_{5,j_2,r}\pm\widetilde{O}(\sigma_0)\Big)\pm\widetilde{O}(\delta^q)\Big)\right.$$

$$-\sum_{j\neq j_2\in\tau(\mathcal{Y})}\mathbf{logit}_{5,j}\cdot\Big(\sum_{r\in\widehat{\mathfrak{A}}_j}\mathbf{sReLU}'(\Lambda_{5,j,r})\cdot\Big(V_{j,r}(g_2)-\Lambda_{5,j,r}\pm\widetilde{O}(\sigma_0)\Big)\pm\widetilde{O}(\delta^q)\Big)$$

$$\left.\pm\sum_{j\notin\tau(\mathcal{Y})}\mathbf{logit}_{5,j}\widetilde{O}(\sigma_0^q)\right)\mathbb{1}_{\tau(x_1)=s}\Bigg].$$

*Moreover, for the off-diagonal entries $[\mathbf{Q}_{4,3}]_{s,s'}$ with $s \neq s'$, we have*

$$\left[-\nabla_{\mathbf{Q}_{4,3}}\mathsf{Loss}_5^{2,1}\right]_{s,s'} = \mathbb{E}\Bigg[\mathbf{Attn}_{\mathsf{ans},0\to\mathsf{pred},2}\cdot$$

$$\left((1 - \mathbf{logit}_{5,j_1}) \cdot \Big( \sum_{r \in \widehat{\mathfrak{A}}_{j_1}} \mathbf{sReLU}'(\Lambda_{5,j_1,r}) \cdot \Big(V_{j_1,r}(g_2) - \Lambda_{5,j_1,r} \pm \widetilde{O}(\sigma_0)\Big) \pm \widetilde{O}(\delta^q)\Big)\right.$$

$$-\sum_{j \neq j_1 \in \tau(\mathcal{Y})} \mathbf{logit}_{5,j} \cdot \Big( \sum_{r \in \widehat{\mathfrak{A}}_j} \mathbf{sReLU}'(\Lambda_{5,j,r}) \cdot \Big(V_{j,r}(g_2) - \Lambda_{5,j,r} \pm \widetilde{O}(\sigma_0)\Big) \pm \widetilde{O}(\delta^q)\Big)$$

$$\left.\pm \sum_{j \notin \tau(\mathcal{Y})} \mathbf{logit}_{5,j}\widetilde{O}(\sigma_0^q)\Big) \mathbb{1}_{\tau(x_0)=s, \tau(x_1)=s'}\right];$$

$$\left[-\nabla_{\mathbf{Q}_{4,3}}\mathsf{Loss}_5^{2,2}\right]_{s,s'} = \mathbb{E}\left[\mathbf{Attn}_{\mathsf{ans},1\to\mathsf{pred},1}\cdot\right.$$

$$\left((1 - \mathbf{logit}_{5,j_2}) \cdot \Big( \sum_{r \in \widehat{\mathfrak{A}}_{j_2}} \mathbf{sReLU}'(\Lambda_{5,j_2,r}) \cdot \Big(V_{j_2,r}(g_1) - \Lambda_{5,j_2,r} \pm \widetilde{O}(\sigma_0)\Big) \pm \widetilde{O}(\delta^q)\Big)\right.$$

$$-\sum_{j \neq j_2 \in \tau(\mathcal{Y})} \mathbf{logit}_{5,j} \cdot \Big( \sum_{r \in \widehat{\mathfrak{A}}_j} \mathbf{sReLU}'(\Lambda_{5,j,r}) \cdot \Big(V_{j,r}(g_1) - \Lambda_{5,j,r} \pm \widetilde{O}(\sigma_0)\Big) \pm \widetilde{O}(\delta^q)\Big)$$

$$\left.\left.\pm \sum_{j \notin \tau(\mathcal{Y})} \mathbf{logit}_{5,j}\widetilde{O}(\sigma_0^q)\Big) \mathbb{1}_{\tau(x_1)=s, \tau(x_0)=s'}\right]\right..$$

*Proof.* For $\ell = 1$,

- for the diagonal entry $[\mathbf{Q}_{4,3}]_{s,s}$ with $s \in \tau(\mathcal{X})$,

$$\left[-\nabla_{\mathbf{Q}_{4,3}}\mathsf{Loss}_5^{2,1}\right]_{s,s} = \mathbb{E}\left[\mathbf{Attn}_{\mathsf{ans},0\to\mathsf{pred},1} \cdot \Big( \sum_{j \in [d]} \mathcal{E}_{5,j}(\mathbf{Z}^{2,0}) \sum_{r \in [m]} \mathbf{sReLU}'(\Lambda_{5,j,r})\cdot\right.$$

$$\left.\Big(\langle \mathbf{W}_{5,j,r}, \mathbf{Z}_{\mathsf{pred},1}\rangle - \Lambda_{5,j,r} + b_{5,j,r}\Big)\Big) \mathbb{1}_{s=\tau(x_0)}\right]$$

$$= \mathbb{E}\left[\mathbf{Attn}_{\mathsf{ans},0\to\mathsf{pred},1} \cdot \Big((1 - \mathbf{logit}_{5,j_1}) \cdot \sum_{r \in [m]} \mathbf{sReLU}'(\Lambda_{5,j_1,r})\Big(V_{j_1,r}(g_1) - \Lambda_{5,j_1,r} \pm \widetilde{O}(\sigma_0)\Big)\right.$$

$$-\sum_{j \neq j_1 \in \tau(\mathcal{Y})} \mathbf{logit}_{5,j} \cdot \sum_{r \in [m]} \mathbf{sReLU}'(\Lambda_{5,j,r})\Big(V_{j,r}(g_1) - \Lambda_{5,j,r} \pm \widetilde{O}(\sigma_0)\Big)$$

$$\left.-\sum_{j \notin \tau(\mathcal{Y})} \mathbf{logit}_{5,j} \cdot \sum_{r \in [m]} \mathbf{sReLU}'(\Lambda_{5,j,r})\Big(\langle\mathbf{W}_{5,j,r}, \mathbf{Z}_{\mathsf{pred},1}\rangle - \Lambda_{5,j,r} \pm \widetilde{O}(\sigma_0)\Big)\Big) \mathbb{1}_{\tau(x_0)=s}\right]$$

$$= \mathbb{E}\left[\mathbf{Attn}_{\mathsf{ans},0\to\mathsf{pred},1}\cdot\right.$$

$$\left((1 - \mathbf{logit}_{5,j_1}) \cdot \Big( \sum_{r \in \widehat{\mathfrak{A}}_{j_1}} \mathbf{sReLU}'(\Lambda_{5,j_1,r}) \cdot \Big(V_{j_1,r}(g_1) - \Lambda_{5,j_1,r} \pm \widetilde{O}(\sigma_0)\Big) \pm \widetilde{O}(\delta^q)\Big)\right.$$

$$-\sum_{j \neq j_1 \in \tau(\mathcal{Y})} \mathbf{logit}_{5,j} \cdot \Big( \sum_{r \in \widehat{\mathfrak{A}}_j} \mathbf{sReLU}'(\Lambda_{5,j,r}) \cdot \Big(V_{j,r}(g_1) - \Lambda_{5,j,r} \pm \widetilde{O}(\sigma_0)\Big) \pm \widetilde{O}(\delta^q)\Big)$$

$$\left.\left.\pm \sum_{j \notin \tau(\mathcal{Y})} \mathbf{logit}_{5,j}\widetilde{O}(\sigma_0^q)\Big) \mathbb{1}_{\tau(x_0)=s}\right],\right.$$

where the last equality follows from Lemma G.6 and Lemma G.7.

- for the non-diagonal entry $[\mathbf{Q}_{4,3}]_{s,s'}$ with $s \neq s' \in \tau(\mathcal{X})$, the analysis is similar unless the condition that the gradient is non-zero only when $s = \tau(x_0)$ and $s' = \tau(x_1)$. Thus, we have

$$\left[ -\nabla_{\mathbf{Q}_{4,3}} \mathsf{Loss}_5^{2,1} \right]_{s,s'} = \mathbb{E}\Bigg[ \mathbf{Attn}_{\mathsf{ans},0\to\mathsf{pred},2} \cdot \Bigg( \sum_{j\in[d]} \mathcal{E}_{5,j}(\mathbf{Z}^{2,0}) \sum_{r\in[m]} \mathbf{sReLU}'(\Lambda_{5,j,r}) \cdot$$

$$\Big( \langle \mathbf{W}_{5,j,r}, \mathbf{Z}_{\mathsf{pred},2} \rangle - \Lambda_{5,j,r} + b_{5,j,r} \Big) \Bigg) \mathbb{1}_{s=\tau(x_0),s'=\tau(x_1)} \Bigg]$$

$$= \mathbb{E}\Bigg[ \mathbf{Attn}_{\mathsf{ans},0\to\mathsf{pred},2} \cdot$$

$$\Bigg( (1 - \mathbf{logit}_{5,j_1}) \cdot \Big( \sum_{r\in\widehat{\mathfrak{A}}_{j_1}} \mathbf{sReLU}'(\Lambda_{5,j_1,r}) \cdot \Big( V_{j_1,r}(g_2) - \Lambda_{5,j_1,r} \pm \widetilde{O}(\sigma_0) \Big) \pm \widetilde{O}(\delta^q) \Big)$$

$$- \sum_{j\neq j_1\in\tau(\mathcal{Y})} \mathbf{logit}_{5,j} \cdot \Big( \sum_{r\in\widehat{\mathfrak{A}}_j} \mathbf{sReLU}'(\Lambda_{5,j,r}) \cdot \Big( V_{j,r}(g_2) - \Lambda_{5,j,r} \pm \widetilde{O}(\sigma_0) \Big) \pm \widetilde{O}(\delta^q) \Big)$$

$$\pm \sum_{j\notin\tau(\mathcal{Y})} \mathbf{logit}_{5,j} \widetilde{O}(\sigma_0^q) \Bigg) \mathbb{1}_{\tau(x_0)=s,\tau(x_1)=s'} \Bigg].$$

For $\ell = 2$,

- for the diagonal entry $[\mathbf{Q}_{4,3}]_{s,s}$ with $s \in \tau(\mathcal{X})$,

$$\left[ -\nabla_{\mathbf{Q}_{4,3}} \mathsf{Loss}_5^{2,2} \right]_{s,s} = \mathbb{E}\Bigg[ \mathbf{Attn}_{\mathsf{ans},1\to\mathsf{pred},2} \cdot \Bigg( \sum_{j\in[d]} \mathcal{E}_{5,j}(\mathbf{Z}^{2,1}) \sum_{r\in[m]} \mathbf{sReLU}'(\Lambda_{5,j,r}) \cdot$$

$$\Big( \langle \mathbf{W}_{5,j,r}, \mathbf{Z}_{\mathsf{pred},2} \rangle - \Lambda_{5,j,r} + b_{5,j,r} \Big) \Bigg) \mathbb{1}_{s=\tau(x_1)} \Bigg]$$

$$= \mathbb{E}\Bigg[ \mathbf{Attn}_{\mathsf{ans},1\to\mathsf{pred},2} \cdot \Bigg( (1 - \mathbf{logit}_{5,j_2}) \cdot \sum_{r\in[m]} \mathbf{sReLU}'(\Lambda_{5,j_2,r}) \Big( V_{j_2,r}(g_2) - \Lambda_{5,j_2,r} \pm \widetilde{O}(\sigma_0) \Big)$$

$$- \sum_{j\neq j_2\in\tau(\mathcal{Y})} \mathbf{logit}_{5,j} \cdot \sum_{r\in[m]} \mathbf{sReLU}'(\Lambda_{5,j,r}) \Big( V_{j,r}(g_2) - \Lambda_{5,j,r} \pm \widetilde{O}(\sigma_0) \Big)$$

$$- \sum_{j\notin\tau(\mathcal{Y})} \mathbf{logit}_{5,j} \cdot \sum_{r\in[m]} \mathbf{sReLU}'(\Lambda_{5,j,r}) \Big( \langle \mathbf{W}_{5,j,r}, \mathbf{Z}_{\mathsf{pred},2} \rangle - \Lambda_{5,j,r} \pm \widetilde{O}(\sigma_0) \Big) \Bigg) \mathbb{1}_{\tau(x_1)=s} \Bigg]$$

$$= \mathbb{E}\Bigg[ \mathbf{Attn}_{\mathsf{ans},1\to\mathsf{pred},2} \cdot$$

$$\Bigg( (1 - \mathbf{logit}_{5,j_2}) \cdot \Big( \sum_{r\in\widehat{\mathfrak{A}}_{j_2}} \mathbf{sReLU}'(\Lambda_{5,j_2,r}) \cdot \Big( V_{j_2,r}(g_2) - \Lambda_{5,j_2,r} \pm \widetilde{O}(\sigma_0) \Big) \pm \widetilde{O}(\delta^q) \Big)$$

$$- \sum_{j\neq j_2\in\tau(\mathcal{Y})} \mathbf{logit}_{5,j} \cdot \Big( \sum_{r\in\widehat{\mathfrak{A}}_j} \mathbf{sReLU}'(\Lambda_{5,j,r}) \cdot \Big( V_{j,r}(g_2) - \Lambda_{5,j,r} \pm \widetilde{O}(\sigma_0) \Big) \pm \widetilde{O}(\delta^q) \Big)$$

$$\pm \sum_{j\notin\tau(\mathcal{Y})} \mathbf{logit}_{5,j} \widetilde{O}(\sigma_0^q) \Bigg) \mathbb{1}_{\tau(x_1)=s} \Bigg],$$

where the last equality follows from Lemma G.6 and Lemma G.7.

- for the non-diagonal entry $[\mathbf{Q}_{4,3}]_{s,s'}$ with $s \neq s' \in \tau(\mathcal{X})$, the analysis is similar unless the condition that the gradient is non-zero only when $s = \tau(x_0)$ and $s' = \tau(x_1)$. Thus, we have

$$
\left[-\nabla_{\mathbf{Q}_{4,3}}\mathsf{Loss}_5^{2,2}\right]_{s,s'} = \mathbb{E}\Bigg[\mathbf{Attn}_{\mathsf{ans},1\to\mathsf{pred},1} \cdot \Bigg(\sum_{j\in[d]}\mathcal{E}_{5,j}(\mathbf{Z}^{2,1})\sum_{r\in[m]}\mathbf{sReLU}'(\Lambda_{5,j,r})\cdot
$$

$$
\Big(\langle\mathbf{W}_{5,j,r},\mathbf{Z}_{\mathsf{pred},1}\rangle - \Lambda_{5,j,r} + b_{5,j,r}\Big)\Bigg)\mathbb{1}_{s=\tau(x_1),s'=\tau(x_0)}\Bigg]
$$

$$
= \mathbb{E}\Bigg[\mathbf{Attn}_{\mathsf{ans},1\to\mathsf{pred},1}\cdot
$$

$$
\Bigg(\Big(1-\mathbf{logit}_{5,j_2}\Big)\cdot\Big(\sum_{r\in\widehat{\mathfrak{A}}_{j_2}}\mathbf{sReLU}'(\Lambda_{5,j_2,r})\cdot\Big(V_{j_2,r}(g_1) - \Lambda_{5,j_2,r}\pm\widetilde{O}(\sigma_0)\Big)\pm\widetilde{O}(\delta^q)\Big)
$$

$$
- \sum_{j\neq j_2\in\tau(\mathcal{Y})}\mathbf{logit}_{5,j}\cdot\Big(\sum_{r\in\widehat{\mathfrak{A}}_{j}}\mathbf{sReLU}'(\Lambda_{5,j,r})\cdot\Big(V_{j,r}(g_1) - \Lambda_{5,j,r}\pm\widetilde{O}(\sigma_0)\Big)\pm\widetilde{O}(\delta^q)\Big)
$$

$$
\pm\sum_{j\notin\tau(\mathcal{Y})}\mathbf{logit}_{5,j}\widetilde{O}(\sigma_0^q)\Bigg)\mathbb{1}_{\tau(x_1)=s,\tau(x_0)=s'}\Bigg].
$$

Therefore, we complete the proof. □

**Lemma G.9** (Refined expression for the gradient of $\mathbf{Q}_{4,4}$). *Given $s \in \tau(\mathcal{X})$, for the diagonal entry* $[\mathbf{Q}_{4,4}]_{s,s}$ *of the block* $\mathbf{Q}_{4,4}$, *letting $j_1 = \tau(g_1(y_0))$ and $j_2 = \tau(g_2(y_1))$, we have*

$$
\left[-\nabla_{\mathbf{Q}_{4,4}}\mathsf{Loss}_5^{2,1}\right]_{s,s} = \mathbb{E}\Bigg[\mathbf{Attn}_{\mathsf{ans},0\to\mathsf{ans},0}\cdot
$$

$$
\Bigg(\Big(1-\mathbf{logit}_{5,j_1}\Big)\cdot\Big(\sum_{r\in\widehat{\mathfrak{A}}_{j_1}}\mathbf{sReLU}'(\Lambda_{5,j_1,r})\cdot\Big(V_{j_1,r}(y_0) - \Lambda_{5,j_1,r}\pm\widetilde{O}(\sigma_0)\Big)\pm\widetilde{O}(\delta^q)\Big)
$$

$$
- \sum_{j\neq j_1\in\tau(\mathcal{Y})}\mathbf{logit}_{5,j}\cdot\Big(\sum_{r\in\widehat{\mathfrak{A}}_{j}}\mathbf{sReLU}'(\Lambda_{5,j,r})\cdot\Big(V_{j,r}(y_0) - \Lambda_{5,j,r}\pm\widetilde{O}(\sigma_0)\Big)\pm\widetilde{O}(\delta^q)\Big)
$$

$$
\pm\sum_{j\notin\tau(\mathcal{Y})}\mathbf{logit}_{5,j}\widetilde{O}(\sigma_0^q)\Bigg)\mathbb{1}_{\tau(x_0)=s}\Bigg].
$$

$$
\left[-\nabla_{\mathbf{Q}_{4,4}}\mathsf{Loss}_5^{2,2}\right]_{s,s} = \mathbb{E}\Bigg[\mathbf{Attn}_{\mathsf{ans},1\to\mathsf{ans},1}\cdot
$$

$$
\Bigg(\Big(1-\mathbf{logit}_{5,j_2}\Big)\cdot\Big(\sum_{r\in\widehat{\mathfrak{A}}_{j_2}}\mathbf{sReLU}'(\Lambda_{5,j_2,r})\cdot\Big(V_{j_2,r}(y_1) - \Lambda_{5,j_2,r}\pm\widetilde{O}(\sigma_0)\Big)\pm\widetilde{O}(\delta^q)\Big)
$$

$$
- \sum_{j\neq j_2\in\tau(\mathcal{Y})}\mathbf{logit}_{5,j}\cdot\Big(\sum_{r\in\widehat{\mathfrak{A}}_{j}}\mathbf{sReLU}'(\Lambda_{5,j,r})\cdot\Big(V_{j,r}(y_1) - \Lambda_{5,j,r}\pm\widetilde{O}(\sigma_0)\Big)\pm\widetilde{O}(\delta^q)\Big)
$$

$$
\pm\sum_{j\notin\tau(\mathcal{Y})}\mathbf{logit}_{5,j}\widetilde{O}(\sigma_0^q)\Bigg)\mathbb{1}_{\tau(x_1)=s}\Bigg].
$$

*Moreover, for the off-diagonal entries $[\mathbf{Q}_{4,4}]_{s,s'}$ with $s \neq s'$, we have $[-\nabla_{\mathbf{Q}_{4,4}}\mathsf{Loss}_5^{2,1}]_{s,s'} = 0$, and*

$$
\left[-\nabla_{\mathbf{Q}_{4,4}}\mathsf{Loss}_5^{2,2}\right]_{s,s'} = \mathbb{E}\Bigg[\mathbf{Attn}_{\mathsf{ans},1\to\mathsf{ans},0}\cdot
$$

$$\left( (1 - \mathbf{logit}_{5,j_2}) \cdot \Big( \sum_{r \in \widehat{\mathfrak{A}}_{j_2}} \mathbf{sReLU}'(\Lambda_{5,j_2,r}) \cdot \Big( V_{j_2,r}(y_0) - \Lambda_{5,j_2,r} \pm \widetilde{O}(\sigma_0) \Big) \pm \widetilde{O}(\delta^q) \Big) \right.$$

$$- \sum_{j \neq j_2 \in \tau(\mathcal{Y})} \mathbf{logit}_{5,j} \cdot \Big( \sum_{r \in \widehat{\mathfrak{A}}_j} \mathbf{sReLU}'(\Lambda_{5,j,r}) \cdot \Big( V_{j,r}(y_0) - \Lambda_{5,j,r} \pm \widetilde{O}(\sigma_0) \Big) \pm \widetilde{O}(\delta^q) \Big)$$

$$\left. \pm \sum_{j \notin \tau(\mathcal{Y})} \mathbf{logit}_{5,j} \widetilde{O}(\sigma_0^q) \Big) \mathbb{1}_{\tau(x_1)=s,\tau(x_0)=s'} \right].$$

The proof is similar to Lemma G.8 and we omit it here.

**Notations for gradient decompositions.** We shall define some useful notations to further simplify the expressions of the gradient.

- for $\ell = 1$,
  - for $[\mathbf{Q}_{4,3}]_{s,s}$ with $s \in \tau(\mathcal{X})$, we have $[-\nabla_{\mathbf{Q}_{4,3}} \mathrm{Loss}_5^{2,1}]_{s,s} = \mathcal{N}_{s,3,1,i} + \mathcal{N}_{s,3,1,ii} + \mathcal{N}_{s,3,1,iii}$, where

$$\mathcal{N}_{s,3,1,i} = \mathbb{E}\left[ \mathbf{Attn}_{\mathsf{ans},0 \to \mathsf{pred},1} \cdot (1 - \mathbf{logit}_{5,j_1}) \cdot \right. \tag{50}$$

$$\left. \Big( \sum_{r \in \widehat{\mathfrak{A}}_{j_1}} \mathbf{sReLU}'(\Lambda_{5,j_1,r}) \cdot \Big( V_{j_1,r}(g_1) - \Lambda_{5,j_1,r} \pm \widetilde{O}(\sigma_0) \Big) \pm \widetilde{O}(\delta^q) \Big) \mathbb{1}_{\tau(x_0)=s} \right],$$

$$\mathcal{N}_{s,3,1,ii} = -\mathbb{E}\left[ \mathbf{Attn}_{\mathsf{ans},0 \to \mathsf{pred},1} \cdot \sum_{j \neq j_1 \in \tau(\mathcal{Y})} \mathbf{logit}_{5,j} \cdot \right. \tag{51}$$

$$\left. \Big( \sum_{r \in \widehat{\mathfrak{A}}_j} \mathbf{sReLU}'(\Lambda_{5,j,r}) \cdot \Big( V_{j,r}(g_1) - \Lambda_{5,j,r} \pm \widetilde{O}(\sigma_0) \Big) \pm \widetilde{O}(\delta^q) \Big) \mathbb{1}_{\tau(x_0)=s} \right],$$

$$\mathcal{N}_{s,3,1,iii} = \pm\mathbb{E}\left[ \mathbf{Attn}_{\mathsf{ans},0 \to \mathsf{pred},1} \cdot \sum_{j \notin \tau(\mathcal{Y})} \mathbf{logit}_{5,j} \widetilde{O}(\sigma_0^q) \mathbb{1}_{\tau(x_0)=s} \right]. \tag{52}$$

  - for $[\mathbf{Q}_{4,4}]_{s,s}$ with $s \in \tau(\mathcal{X})$, we have $[-\nabla_{\mathbf{Q}_{4,4}} \mathrm{Loss}_5^{2,1}]_{s,s} = \mathcal{N}_{s,4,1,i} + \mathcal{N}_{s,4,1,ii} + \mathcal{N}_{s,4,1,iii}$, where

$$\mathcal{N}_{s,4,1,i} = \mathbb{E}\left[ \mathbf{Attn}_{\mathsf{ans},0 \to \mathsf{ans},0} \cdot (1 - \mathbf{logit}_{5,j_1}) \cdot \right. \tag{53}$$

$$\left. \Big( \sum_{r \in \widehat{\mathfrak{A}}_{j_1}} \mathbf{sReLU}'(\Lambda_{5,j_1,r}) \cdot \Big( V_{j_1,r}(y_0) - \Lambda_{5,j_1,r} \pm \widetilde{O}(\sigma_0) \Big) \pm \widetilde{O}(\delta^q) \Big) \mathbb{1}_{\tau(x_0)=s} \right],$$

$$\mathcal{N}_{s,4,1,ii} = -\mathbb{E}\left[ \mathbf{Attn}_{\mathsf{ans},0 \to \mathsf{ans},0} \cdot \sum_{j \neq j_1 \in \tau(\mathcal{Y})} \mathbf{logit}_{5,j} \cdot \right. \tag{54}$$

$$\left. \Big( \sum_{r \in \widehat{\mathfrak{A}}_j} \mathbf{sReLU}'(\Lambda_{5,j,r}) \cdot \Big( V_{j,r}(y_0) - \Lambda_{5,j,r} \pm \widetilde{O}(\sigma_0) \Big) \pm \widetilde{O}(\delta^q) \Big) \mathbb{1}_{\tau(x_0)=s} \right],$$

$$\mathcal{N}_{s,4,1,iii} = \pm\mathbb{E}\left[ \mathbf{Attn}_{\mathsf{ans},0 \to \mathsf{ans},0} \cdot \sum_{j \notin \tau(\mathcal{Y})} \mathbf{logit}_{5,j} \widetilde{O}(\sigma_0^q) \mathbb{1}_{\tau(x_0)=s} \right]. \tag{55}$$

- for $\ell = 2$,

- for $[\mathbf{Q}_{4,3}]_{s,s}$ with $s \in \tau(\mathcal{X})$, we have $[-\nabla_{\mathbf{Q}_{4,3}}\mathsf{Loss}_5^{2,2}]_{s,s} = \mathcal{N}_{s,3,2,i} + \mathcal{N}_{s,3,2,ii} + \mathcal{N}_{s,3,2,iii}$, where

$$\mathcal{N}_{s,3,2,i} = \mathbb{E}\Bigg[\mathbf{Attn}_{\mathsf{ans},1\to\mathsf{pred},2} \cdot (1 - \mathbf{logit}_{5,j_2}) \cdot \tag{56}$$
$$\Bigg(\sum_{r\in\widehat{\mathfrak{A}}_{j_2}} \mathbf{sReLU}'(\Lambda_{5,j_2,r}) \cdot \Big(V_{j_2,r}(g_2) - \Lambda_{5,j_2,r} \pm \widetilde{O}(\sigma_0)\Big) \pm \widetilde{O}(\delta^q)\Bigg)\mathbb{1}_{\tau(x_1)=s}\Bigg],$$

$$\mathcal{N}_{s,3,2,ii} = -\mathbb{E}\Bigg[\mathbf{Attn}_{\mathsf{ans},1\to\mathsf{pred},2} \cdot \sum_{j\neq j_2\in\tau(\mathcal{Y})} \mathbf{logit}_{5,j} \cdot \tag{57}$$
$$\Bigg(\sum_{r\in\widehat{\mathfrak{A}}_{j}} \mathbf{sReLU}'(\Lambda_{5,j,r}) \cdot \Big(V_{j,r}(g_2) - \Lambda_{5,j,r} \pm \widetilde{O}(\sigma_0)\Big) \pm \widetilde{O}(\delta^q)\Bigg)\mathbb{1}_{\tau(x_1)=s}\Bigg],$$

$$\mathcal{N}_{s,3,2,iii} = \pm\mathbb{E}\Bigg[\mathbf{Attn}_{\mathsf{ans},1\to\mathsf{pred},2} \cdot \sum_{j\notin\tau(\mathcal{Y})} \mathbf{logit}_{5,j}\widetilde{O}(\sigma_0^q)\mathbb{1}_{\tau(x_1)=s}\Bigg]. \tag{58}$$

- for $[\mathbf{Q}_{4,4}]_{s,s}$ with $s \in \tau(\mathcal{X})$, we have $[-\nabla_{\mathbf{Q}_{4,4}}\mathsf{Loss}_5^{2,2}]_{s,s} = \mathcal{N}_{s,4,2,i} + \mathcal{N}_{s,4,2,ii} + \mathcal{N}_{s,4,2,iii}$, where

$$\mathcal{N}_{s,4,2,i} = \mathbb{E}\Bigg[\mathbf{Attn}_{\mathsf{ans},1\to\mathsf{ans},1} \cdot (1 - \mathbf{logit}_{5,j_2}) \cdot \tag{59}$$
$$\Bigg(\sum_{r\in\widehat{\mathfrak{A}}_{j_2}} \mathbf{sReLU}'(\Lambda_{5,j_2,r}) \cdot \Big(V_{j_2,r}(y_1) - \Lambda_{5,j_2,r} \pm \widetilde{O}(\sigma_0)\Big) \pm \widetilde{O}(\delta^q)\Bigg)\mathbb{1}_{\tau(x_1)=s}\Bigg],$$

$$\mathcal{N}_{s,4,2,ii} = -\mathbb{E}\Bigg[\mathbf{Attn}_{\mathsf{ans},1\to\mathsf{ans},1} \cdot \sum_{j\neq j_2\in\tau(\mathcal{Y})} \mathbf{logit}_{5,j} \cdot \tag{60}$$
$$\Bigg(\sum_{r\in\widehat{\mathfrak{A}}_{j}} \mathbf{sReLU}'(\Lambda_{5,j,r}) \cdot \Big(V_{j,r}(y_1) - \Lambda_{5,j,r} \pm \widetilde{O}(\sigma_0)\Big) \pm \widetilde{O}(\delta^q)\Bigg)\mathbb{1}_{\tau(x_1)=s}\Bigg],$$

$$\mathcal{N}_{s,4,2,iii} = \pm\mathbb{E}\Bigg[\mathbf{Attn}_{\mathsf{ans},1\to\mathsf{ans},1} \cdot \sum_{j\notin\tau(\mathcal{Y})} \mathbf{logit}_{5,j}\widetilde{O}(\sigma_0^q)\mathbb{1}_{\tau(x_1)=s}\Bigg]. \tag{61}$$

**Probabilistic Events.** We conclude this subsection by introducing several probabilistic events that will be used to simplify the characterization of activated neurons in the subsequent analysis.

$$\mathcal{E}_1 \triangleq \Big\{g_1 \neq g_2\Big\}, \tag{62}$$

$$\mathcal{E}_2 \triangleq \Big\{g_{\ell_1}(y_{\ell_1'}) \neq g_{\ell_2}(y_{\ell_2'}), \text{ for any } (\ell_1, \ell_1') \neq (\ell_2, \ell_2'), \text{ where } \ell_k \in [2], \ell_k' \in \{0,1\}\Big\}. \tag{63}$$

It is easy to see that $\mathcal{E}_1$ and $\mathcal{E}_2$ hold with high probability $1 - O\big(\frac{1}{\log d}\big)$.

### G.3 Stage 2.1: Initial Growth of Gap

At the beginning of stage 2, since $\mathbf{Q}$ has not been trained for long, we have the attention score is still close to the uniform structure. Therefore, for $\ell = 1$, we have

$$\Lambda_{5,j,r}(\mathbf{Z}^{2,\ell-1}) \approx \frac{1}{3}V_{j,r}(g_1) + \frac{1}{3}V_{j,r}(g_2) + \frac{1}{3}V_{j,r}(y_0) \pm \widetilde{O}(\sigma_0).$$

If $\mathbf{Z}^{2,\ell-1} \in \mathcal{E}_1$,

- for $j = j_1$, for $r \in \widehat{\mathfrak{A}}_{j_1}$, only $r_{g_1 \cdot y_0}$ is activated since $\Lambda_{5,j_1,r_{g_1 \cdot y_0}} \approx \frac{1}{3}B$ and $\Lambda_{5,j_1,r_{g_2 \cdot g_2^{-1}(j_1)}} \approx -\frac{1}{3}B \ll -\varrho$;

- for $j = j_1' \triangleq \tau(g_2(y_0))$, only $r_{g_2 \cdot y_0}$ is activated since $\Lambda_{5,j_1',r_{g_2 \cdot y_0}} \approx \frac{1}{3}B$ and $\Lambda_{5,j_1',r_{g_1 \cdot g_2^{-1}(j_1')}} \approx -\frac{1}{3}B \ll -\varrho$;

- for other $j \in \tau(\mathcal{Y})$, we have $\Lambda_{5,j,r} \leq -\frac{1}{3}B$ for all $r \in \widehat{\mathfrak{A}}_j$, thus no activation.

Moreover, for $\ell = 2$, we have

$$\Lambda_{5,j,r}(\mathbf{Z}^{2,\ell-1}) \approx \frac{1}{4}V_{j,r}(g_1) + \frac{1}{4}V_{j,r}(g_2) + \frac{1}{4}V_{j,r}(y_0) + \frac{1}{4}V_{j,r}(y_1) \pm \widetilde{O}(\sigma_0).$$

If $\mathbf{Z}^{2,\ell-1} \in \mathcal{E}_2$,

- for $j \in \{\tau(g_\ell(y_{\ell'}))\}_{\ell \in [2], \ell' \in \{0,1\}}$, only the corresponding $r_{g_\ell \cdot y_{\ell'}}$ is activated in the smoothed regime since $|\Lambda_{5,j,r_{g_\ell \cdot y_{\ell'}}}| = O(\delta)$ and $\Lambda_{5,j,r} \approx -\frac{1}{2}B \ll -\varrho$ for $r \in \widehat{\mathfrak{A}}_j \setminus \{r_{g_\ell \cdot y_{\ell'}}\}$;

- for other $j \in \tau(\mathcal{Y})$, we have $\Lambda_{5,j,r} \leq -\frac{1}{2}B$ for all $r \in \widehat{\mathfrak{A}}_j$, thus no activation.

Here, activation means that the corresponding $\mathbf{sReLU}'(\Lambda_{5,j,r})$ is non-zero, which is crucial for the gradient computation. Based on the above observations, we can see that the gradient from $\ell = 2$ is relatively small since $\Lambda$ is only activated in the smoothed regime. Thus, initially, the learning process is dominated by $\nabla_{\mathbf{Q}}\mathsf{Loss}_5^{2,1}$. Moreover, if we take a closer look at the gradient from $\ell = 1$, we have

$$\mathcal{N}_{s,3,1,ii} \approx -\mathbb{E}\Bigg[\mathbf{Attn}_{\mathsf{ans},0 \to \mathsf{pred},1} \cdot \mathbf{logit}_{5,j_1'} \cdot$$

$$\left(\mathbf{sReLU}'(\Lambda_{5,j_1',r_{g_2 \cdot y_0}}) \cdot \left(V_{j_1',r_{g_2 \cdot y_0}}(g_1) - \Lambda_{5,j_1',r_{g_2 \cdot y_0}} \pm \widetilde{O}(\sigma_0)\right) \pm \widetilde{O}(\delta^q)\right)\mathbb{1}_{\tau(x_0)=s}\Bigg] \geq 0,$$

$$\mathcal{N}_{s,4,1,ii} \approx -\mathbb{E}\Bigg[\mathbf{Attn}_{\mathsf{ans},0 \to \mathsf{ans},0} \cdot \mathbf{logit}_{5,j_1'} \cdot$$

$$\left(\mathbf{sReLU}'(\Lambda_{5,j_1',r_{g_2 \cdot y_0}}) \cdot \left(V_{j_1',r_{g_2 \cdot y_0}}(y_0) - \Lambda_{5,j_1',r_{g_2 \cdot y_0}} \pm \widetilde{O}(\sigma_0)\right) \pm \widetilde{O}(\delta^q)\right)\Bigg] \leq 0,$$

since $V_{j_1',r_{g_2 \cdot y_0}}(y_0) \geq \Omega(B)$ while $V_{j_1',r_{g_2 \cdot y_0}}(g_1) \leq -\Omega(B)$. Thus, $[\mathbf{Q}_{4,3}]_{s,s}$ will have a significant positive gradient while $[\mathbf{Q}_{4,4}]_{s,s}$ will have a negative counterpart. This will lead to the growth of the gap between $[\mathbf{Q}_{4,3}]_{s,s}$ and $[\mathbf{Q}_{4,4}]_{s,s}$.

We formally characterize this growth behavior within this substage. At the beginning of each substage, we establish an induction hypothesis that we expect to hold throughout. Subsequently, we analyze the dynamics under this hypothesis within the substage, aiming to prove its validity by the end of sub-stage. Due to the symmetry of $[\mathbf{Q}_{4,3}]_{s,s}$ and $[\mathbf{Q}_{4,4}]_{s,s}$ across $s \in \tau(\mathcal{X})$, we may, without loss of generality, focus on a particular $s \in \tau(\mathcal{X})$.

**Induction G.1.** *Given $s \in \tau(\mathcal{X})$, let $T_{2,1,s}$ denote the first time that $[\mathbf{Q}_{4,3}^{(t)}]_{s,s}$ reaches $\Omega\left(\frac{1}{\log d}\right)$. For all iterations $t < T_{2,1,s}$, we have the following holds*

(a) $[\mathbf{Q}_{4,3}^{(t)}]_{s,s}$ *monotonically increases;*

(b) $\left|[\mathbf{Q}_{4,4}^{(t)}]_{s,s}\right| \leq [\mathbf{Q}_{4,3}^{(t)}]_{s,s}$ *and* $[\mathbf{Q}_{4,3}^{(t)}]_{s,s} - [\mathbf{Q}_{4,4}^{(t)}]_{s,s} = \Theta\left([\mathbf{Q}_{4,3}^{(t)}]_{s,s}\right)$;

(c) *for $p \in \{3,4\}$, for $s' \in \tau(\mathcal{X}) \neq s$, $\left|[\mathbf{Q}_{4,p}^{(t)}]_{s,s'}\right| \leq O\left(\frac{[\mathbf{Q}_{4,p}^{(t)}]_{s,s}}{d}\right)$.*

### G.3.1 Attention and Logit Preliminaries

We first introduce several properties of the attention scores and logits if Induction G.1 holds.

**Lemma G.10.** *If Induction G.1 holds for all iterations $< t$, given input $\mathbf{Z}^{2,\ell-1}$, then we have*

1. *for $\ell = 1$,*

(a) $\mathbf{Attn}^{(t)}_{\mathsf{ans},0\to\mathsf{pred},1} \in \left[\frac{1}{3}, \frac{1}{3} + O\left(\frac{1}{\log d}\right)\right]$;

(b) $\mathbf{Attn}^{(t)}_{\mathsf{ans},0\to\mathsf{pred},2}, \mathbf{Attn}^{(t)}_{\mathsf{ans},0\to\mathsf{ans},0} \leq \mathbf{Attn}^{(t)}_{\mathsf{ans},0\to\mathsf{pred},1}$;

(c) $\left|\mathbf{Attn}^{(t)}_{\mathsf{ans},0\to\mathbf{k}} - \mathbf{Attn}^{(t)}_{\mathsf{ans},0\to\mathbf{k}'}\right| \leq O\left(\frac{1}{\log d}\right)$ for $\mathbf{k} \neq \mathbf{k}' \in \mathcal{I}^{(2,0)}$.

2. *for $\ell = 2$,*

   (a) $\mathbf{Attn}^{(t)}_{\mathsf{ans},1\to\mathsf{pred},2} \in \left[\frac{1}{4}, \frac{1}{4} + O\left(\frac{1}{\log d}\right)\right]$;

   (b) $\mathbf{Attn}^{(t)}_{\mathsf{ans},1\to\mathbf{k}} \leq \mathbf{Attn}^{(t)}_{\mathsf{ans},1\to\mathsf{pred},2}$ *for* $\mathbf{k} \neq (\mathsf{pred}, 2)$;

   (c) *for* $\mathbf{k} \in \{(\mathsf{ans},0), (\mathsf{pred},1)\}$, $\mathbf{k}' \in \{(\mathsf{ans},1), (\mathsf{pred},2)\}$,

$$\left|\mathbf{Attn}^{(t)}_{\mathsf{ans},1\to\mathbf{k}} - \mathbf{Attn}^{(t)}_{\mathsf{ans},1\to\mathbf{k}'}\right| \leq O\left(\frac{1}{\log d}\right);$$

   (d) $\left|\mathbf{Attn}^{(t)}_{\mathsf{ans},1\to\mathsf{pred},1} - \mathbf{Attn}^{(t)}_{\mathsf{ans},1\to\mathsf{ans},0}\right| \leq O\left(\frac{1}{d}\right)$.

*Proof.* For $\ell = 1$, given $\mathbf{Z}^{2,\ell-1}$, according to Assumption C.2, we have

$$\mathbf{Attn}^{(t)}_{\mathsf{ans},0\to\mathsf{pred},1} = \frac{e^{[\mathbf{Q}^{(t)}_{4,3}]_{\tau(x_0),\tau(x_0)}}}{e^{[\mathbf{Q}^{(t)}_{4,3}]_{\tau(x_0),\tau(x_0)}} + e^{[\mathbf{Q}^{(t)}_{4,3}]_{\tau(x_0),\tau(x_1)}} + e^{[\mathbf{Q}^{(t)}_{4,4}]_{\tau(x_0),\tau(x_0)}}}$$

$$= \frac{1}{1 + e^{[\mathbf{Q}^{(t)}_{4,3}]_{\tau(x_0),\tau(x_1)} - [\mathbf{Q}^{(t)}_{4,3}]_{\tau(x_0),\tau(x_0)}} + e^{[\mathbf{Q}^{(t)}_{4,4}]_{\tau(x_0),\tau(x_0)} - [\mathbf{Q}^{(t)}_{4,3}]_{\tau(x_0),\tau(x_0)}}}.$$

Thus, by Induction G.1,

$$-O\left(\frac{1}{\log d}\right) \leq [\mathbf{Q}^{(t)}_{4,3}]_{\tau(x_0),\tau(x_1)} - [\mathbf{Q}^{(t)}_{4,3}]_{\tau(x_0),\tau(x_0)}, [\mathbf{Q}^{(t)}_{4,4}]_{\tau(x_0),\tau(x_0)} - [\mathbf{Q}^{(t)}_{4,3}]_{\tau(x_0),\tau(x_0)} \leq 0,$$

which implies that $0 \leq \mathbf{Attn}^{(t)}_{\mathsf{ans},0\to\mathsf{pred},1} \leq \frac{1}{3} + O\left(\frac{1}{\log d}\right)$. (b) is straightforward since

$$[\mathbf{Q}^{(t)}_{4,3}]_{\tau(x_0),\tau(x_1)}, [\mathbf{Q}^{(t)}_{4,4}]_{\tau(x_0),\tau(x_0)} \leq [\mathbf{Q}^{(t)}_{4,3}]_{\tau(x_0),\tau(x_0)}.$$

(c) is a direct consequence of (a) and (b).

For $\ell = 2$, given $\mathbf{Z}^{2,\ell-1}$, (a)- (c) are very similar to the above analysis, and then for (d), we have

$$\mathbf{Attn}^{(t)}_{\mathsf{ans},1\to\mathsf{pred},1} - \mathbf{Attn}^{(t)}_{\mathsf{ans},1\to\mathsf{ans},0}$$

$$= \frac{e^{[\mathbf{Q}^{(t)}_{4,3}]_{\tau(x_1),\tau(x_0)}} - e^{[\mathbf{Q}^{(t)}_{4,4}]_{\tau(x_1),\tau(x_0)}}}{e^{[\mathbf{Q}^{(t)}_{4,3}]_{\tau(x_1),\tau(x_0)}} + e^{[\mathbf{Q}^{(t)}_{4,3}]_{\tau(x_1),\tau(x_1)}} + e^{[\mathbf{Q}^{(t)}_{4,4}]_{\tau(x_1),\tau(x_0)}} + e^{[\mathbf{Q}^{(t)}_{4,4}]_{\tau(x_1),\tau(x_1)}}}$$

$$\overset{(i)}{\leq} O\left([\mathbf{Q}^{(t)}_{4,3}]_{\tau(x_1),\tau(x_0)} - [\mathbf{Q}^{(t)}_{4,4}]_{\tau(x_1),\tau(x_0)}\right) \overset{(ii)}{\leq} O\left(\frac{1}{d}\right),$$

where (i) is due to the fact that $|e^x - e^y| \leq O(|x - y|)$ when $x, y$ are small, and (ii) is due to Induction G.1 (b). $\qquad\square$

**Lemma G.11.** *If Induction G.1 holds for all iterations $< t$, given input $\mathbf{Z}^{2,\ell-1}$, then we have*

1. *for $\ell = 1$, if $\mathbf{Z}^{2,\ell-1} \in \mathcal{E}_1$, then*

   (a) *for $j = j_1$, $\Lambda^{(t)}_{5,j_1,r} \ll -\varrho$ for $r \in \widehat{\mathfrak{A}}_{j_1} \setminus \{r_{g_1 \cdot y_0}\}$;*

   (b) *for $j = j'_1 \triangleq \tau(g_2(y_0))$, $\Lambda^{(t)}_{5,j'_1,r} \ll -\varrho$ for $r \in \widehat{\mathfrak{A}}_{j'_1} \setminus \{r_{g_2 \cdot y_0}\}$;*

   (c) *for other $j \in \tau(\mathcal{Y})$, $r$ is not activated for all $r \in \widehat{\mathfrak{A}}_j$, i.e., $\Lambda^{(t)}_{5,j,r} \ll -\varrho$.*

2. *$\ell = 2$, if $\mathbf{Z}^{2,\ell-1} \in \mathcal{E}_2$, then*

   (a) *for $j \in \{\tau(g_\ell(y_{\ell'}))\}_{\ell \in [2], \ell' \in \{0,1\}}$, only the corresponding $r_{g_\ell \cdot y_{\ell'}}$ may be activated, with $|\Lambda^{(t)}_{5,j,r_{g_\ell \cdot y_{\ell'}}}| \leq O(1)$, while all other $\Lambda^{(t)}_{5,j,r} \ll -\varrho$ for $r \in \widehat{\mathfrak{A}}_j \setminus \{r_{g_\ell \cdot y_{\ell'}}\}$;*

   (b) *for other $j \in \tau(\mathcal{Y})$, $r$ is not activated for all $r \in \widehat{\mathfrak{A}}_j$, i.e., $\Lambda^{(t)}_{5,j,r} \ll -\varrho$.*

*Proof.* We only prove (a) for $\ell = 2$ since the other cases are straightforward. $j = \tau(g_\ell(y_{\ell'}))$, by Lemma G.6 we have

$$\Lambda_{5,j,r_{g_\ell \cdot y_{\ell'}}} = \mathbf{Attn}^{(t)}_{\mathsf{ans},1\to\mathsf{pred},1} V_{j,r_{g_\ell \cdot y_{\ell'}}}(g_1) + \mathbf{Attn}^{(t)}_{\mathsf{ans},1\to\mathsf{pred},2} V_{j,r_{g_\ell \cdot y_{\ell'}}}(g_2)$$

$$+ \mathbf{Attn}^{(t)}_{\mathsf{ans},1\to\mathsf{ans},0} V_{j,r_{g_\ell \cdot y_{\ell'}}}(y_0) + \mathbf{Attn}^{(t)}_{\mathsf{ans},1\to\mathsf{ans},1} V_{j,r_{g_\ell \cdot y_{\ell'}}}(y_1) \pm \widetilde{O}(\sigma_0).$$

Notice that since $\mathbf{Z}^{2,\ell-1} \in \mathcal{E}_2$, we have two $V$ terms are positive and two are negative with magnitude $B \pm O(\delta)$. Therefore, $|\Lambda_{5,j,r_{g_\ell \cdot y_{\ell'}}}| \leq O\left(\frac{1}{\log d}\right) \cdot B = O(1)$. $\qquad\square$

**Lemma G.12.** *If Induction G.1 holds for all iterations $< t$, given input $\mathbf{Z}^{2,0} \in \mathcal{E}_1$, then we have*

*(a) $\Lambda^{(t)}_{5,j_1,r_{g_1 \cdot y_0}} \geq \frac{1}{3}B - O(1)$;*

*(b) $-O(\delta) \leq \Lambda^{(t)}_{5,j_1,r_{g_1 \cdot y_0}} - \Lambda^{(t)}_{5,j_1',r_{g_2 \cdot y_0}} \leq O(1)$;*

*Proof.* By Lemma G.6 we have

$$\Lambda_{5,j_1,r_{g_1 \cdot y_0}} =$$
$$\mathbf{Attn}^{(t)}_{\mathsf{ans},0\to\mathsf{pred},1} V_{j_1,r_{g_1 \cdot y_0}}(g_1) + \mathbf{Attn}^{(t)}_{\mathsf{ans},0\to\mathsf{pred},2} V_{j_1,r_{g_1 \cdot y_0}}(g_2) + \mathbf{Attn}^{(t)}_{\mathsf{ans},0\to\mathsf{ans},0} V_{j,r_{g_1 \cdot y_0}}(y_0) \pm \widetilde{O}(\sigma_0).$$

By Lemma G.10 and the cancellation in (44), we have

$$\mathbf{Attn}^{(t)}_{\mathsf{ans},0\to\mathsf{pred},2} V_{j_1,r_{g_1 \cdot y_0}}(g_2) + \mathbf{Attn}^{(t)}_{\mathsf{ans},0\to\mathsf{ans},0} V_{j_1,r_{g_1 \cdot y_0}}(y_0)$$

$$\geq \left(\frac{1}{3} - O\left(\frac{1}{\log d}\right)\right) \cdot O(\delta) - O\left(\frac{1}{\log d}\right) \cdot B.$$

Putting it back, and using the fact that $\mathbf{Attn}^{(t)}_{\mathsf{ans},0\to\mathsf{pred},1} V_{j_1,r_{g_1 \cdot y_0}}(g_1) \geq \frac{1}{3} \cdot (B - O(\delta))$, we have

$$\Lambda_{5,j_1,r_{g_1 \cdot y_0}} \geq \frac{1}{3} \cdot (B - O(\delta)) + \left(\frac{1}{3} - O\left(\frac{1}{\log d}\right)\right) \cdot O(\delta) - O\left(\frac{1}{\log d}\right) \cdot B \pm \widetilde{O}(\sigma_0)$$

$$\geq \frac{1}{3}B - O(1).$$

Moving on to (b), we have

$$\Lambda_{5,j_1,r_{g_1 \cdot y_0}} - \Lambda_{5,j_1',r_{g_2 \cdot y_0}} = \mathbf{Attn}^{(t)}_{\mathsf{ans},0\to\mathsf{pred},1}(V_{j_1,r_{g_1 \cdot y_0}}(g_1) - V_{j_1',r_{g_2 \cdot y_0}}(g_1)) \pm \widetilde{O}(\sigma_0)$$

$$+ \mathbf{Attn}^{(t)}_{\mathsf{ans},0\to\mathsf{pred},2}(V_{j_1,r_{g_1 \cdot y_0}}(g_2) - V_{j_1',r_{g_2 \cdot y_0}}(g_2)) + \mathbf{Attn}^{(t)}_{\mathsf{ans},0\to\mathsf{ans},0}(V_{j_1,r_{g_1 \cdot y_0}}(y_0) - V_{j_1',r_{g_2 \cdot y_0}}(y_0))$$

$$\leq \mathbf{Attn}^{(t)}_{\mathsf{ans},0\to\mathsf{pred},1} \cdot (2B + O(\delta)) - \mathbf{Attn}^{(t)}_{\mathsf{ans},0\to\mathsf{pred},2} \cdot (2B - O(\delta)) + \mathbf{Attn}^{(t)}_{\mathsf{ans},0\to\mathsf{ans},0} \cdot O(\delta)$$

$$\leq (2B - O(\delta)) \cdot O\left(\frac{1}{\log d}\right) + O(\delta) \leq O(1).$$

Similarly, we have

$$\Lambda_{5,j_1,r_{g_1 \cdot y_0}} - \Lambda_{5,j_1',r_{g_2 \cdot y_0}}$$

$$\geq \mathbf{Attn}^{(t)}_{\mathsf{ans},0\to\mathsf{pred},1} \cdot (2B - O(\delta)) - \mathbf{Attn}^{(t)}_{\mathsf{ans},0\to\mathsf{pred},2} \cdot (2B + O(\delta)) - \mathbf{Attn}^{(t)}_{\mathsf{ans},0\to\mathsf{ans},0} \cdot O(\delta)$$

$$\geq -\mathbf{Attn}^{(t)}_{\mathsf{ans},0\to\mathsf{pred},2} \cdot O(\delta) - \mathbf{Attn}^{(t)}_{\mathsf{ans},0\to\mathsf{ans},0} \cdot O(\delta) \geq -O(\delta).$$

$\qquad\square$

**Lemma G.13.** *If Induction G.1 holds for all iterations $< t$, given input $\mathbf{Z}^{2,\ell-1}$, then we have*

*1. for $\ell = 1$, if $\mathbf{Z}^{2,\ell-1} \in \mathcal{E}_1$, $\mathbf{logit}^{(t)}_{5,j} = \Omega(1)$ for $j \in \{j_1, j_1'\}$, $1 - \mathbf{logit}^{(t)}_{5,j_1} - \mathbf{logit}^{(t)}_{5,j_1'} = \frac{1}{\mathrm{poly}d}$.*

*2. for $\ell = 2$, if $\mathbf{Z}^{2,\ell-1} \in \mathcal{E}_2$, $\mathbf{logit}^{(t)}_{5,j} = O(\frac{1}{d})$ for all $j$.*

*Proof.* • For $\ell = 1$, by Lemma G.11 and Lemma G.6, we have

$$F_{5,j_1}^{(t)}(\mathbf{Z}^{2,\ell-1}) = \sum_{r\in[m]} \mathbf{sReLU}\big(\Lambda_{5,j_1,r}^{(t)}\big) = \Lambda_{5,j_1,r_{g_1\cdot y_0}}^{(t)} + \varrho(m/q - 1) = \Lambda_{5,j_1,r_{g_1\cdot y_0}}^{(t)} + O\Big(\frac{1}{\mathrm{polylog}d}\Big),$$

$$F_{5,j_1'}^{(t)}(\mathbf{Z}^{2,\ell-1}) = \sum_{r\in[m]} \mathbf{sReLU}\big(\Lambda_{5,j_1',r}^{(t)}\big) = \Lambda_{5,j_1',r_{g_2\cdot y_0}}^{(t)} + \varrho(m/q - 1),$$

$$F_{5,j}^{(t)}(\mathbf{Z}^{2,\ell-1}) = \sum_{r\in[m]} \mathbf{sReLU}\big(\Lambda_{5,j,r}^{(t)}\big) \leq O\Big(\frac{1}{\mathrm{polylog}d}\Big) \text{ for } j \neq j_1, j_1' \in \tau(\mathcal{Y}),$$

$$F_{5,j}^{(t)}(\mathbf{Z}^{2,\ell-1}) \leq m \cdot \widetilde{O}(\sigma_0^q) \text{ for } j \notin \tau(\mathcal{Y}).$$

Putting it together, we obtain

$$\mathbf{logit}_{5,j_1}^{(t)} = \frac{1}{1 + e^{F_{5,j_1'}^{(t)} - F_{5,j_1}^{(t)}} + \Big(\sum_{j\neq j_1, j_1'\in\tau(\mathcal{Y})} e^{F_{5,j}^{(t)}} + \sum_{j\notin\tau(\mathcal{Y})} e^{F_{5,j}^{(t)}}\Big) \cdot e^{-F_{5,j_1}^{(t)}}}$$

$$= \frac{1}{1 + e^{\Lambda_{5,j_1',r_{g_2\cdot y_0}}^{(t)} - \Lambda_{5,j_1,r_{g_1\cdot y_0}}^{(t)}} + \Big(O(\log d) \cdot e^{O(\frac{1}{\mathrm{polylog}d})} + O(d) \cdot e^{\widetilde{O}(m\sigma_0^q)}\Big) \cdot e^{-F_{5,j_1}^{(t)}}}.$$

Thus, by Lemma G.12, and the fact that $B = C_B \log d$ for some sufficiently large constant $C_B > 0$, we have $\mathbf{logit}_{5,j_1}^{(t)} = \frac{1}{1 + e^{-O(\delta)} + O(1)\cdot e^{-(C_B/3-1)\log d}} = \Omega(1)$. Similarly, we have

$$\mathbf{logit}_{5,j_1'}^{(t)} = \frac{1}{1 + e^{-\Lambda_{5,j_1',r_{g_2\cdot y_0}}^{(t)} + \Lambda_{5,j_1,r_{g_1\cdot y_0}}^{(t)}} + \Big(O(\log d) \cdot e^{O(\frac{1}{\mathrm{polylog}d})} + O(d) \cdot e^{\widetilde{O}(m\sigma_0^q)}\Big) \cdot e^{-F_{5,j_1'}^{(t)}}}$$

$$= \frac{1}{1 + e^{O(1)} + O(1) \cdot e^{-(C_B/3-1)\log d}} = \Omega(1).$$

From the above analysis, it is easy to see that $1 - \mathbf{logit}_{5,j_1}^{(t)} - \mathbf{logit}_{5,j_1'}^{(t)} \leq O\Big(\frac{1}{e^{(C_B/3-1)\log d}}\Big) = O(\frac{1}{\mathrm{poly}d})$.

• For $\ell = 2$, by Lemma G.11 and Lemma G.6, we have

$$F_{5,j}^{(t)}(\mathbf{Z}^{2,\ell-1}) = \sum_{r\in[m]} \mathbf{sReLU}\big(\Lambda_{5,j_1,r}^{(t)}\big) \in [\varrho m/q, O(1) + \varrho(m/q - 1)] \text{ for } j \in \{\tau(g_\ell(y_{\ell'}))\}_{\ell\in[2],\ell'\in\{0,1\}}$$

$$F_{5,j}^{(t)}(\mathbf{Z}^{2,\ell-1}) = \sum_{r\in[m]} \mathbf{sReLU}\big(\Lambda_{5,j,r}^{(t)}\big) \leq O\Big(\frac{1}{\mathrm{polylog}d}\Big) \text{ for } j \in \tau(\mathcal{Y}) \setminus \{\tau(g_\ell(y_{\ell'}))\}_{\ell\in[2],\ell'\in\{0,1\}}$$

$$F_{5,j}^{(t)}(\mathbf{Z}^{2,\ell-1}) \leq m \cdot \widetilde{O}(\sigma_0^q) \text{ for } j \notin \tau(\mathcal{Y}).$$

Therefore, for any $j$, we have $\mathbf{logit}_j^{(t)} = O\big(\frac{1}{d}\big)$ since $F_{5,j'}^{(t)} \leq O(1)$ for all $j'$. $\qquad\square$

In the following, we illustrate the activations on the non-high probability event.

**Lemma G.14.** *If Induction G.1 holds for all iterations $< t$, given input $\mathbf{Z}^{2,\ell-1}$, then we have*

1. *for $\ell = 1$, if $\mathbf{Z}^{2,\ell-1} \notin \mathcal{E}_1$, then*

   (a) *for $j = j_1$, $\widehat{\mathfrak{A}}_{j_1} = \{r_{g_1\cdot y_0}\}$, $\Lambda_{5,j_1,r_{g_1\cdot y_0}}^{(t)} = B \pm O(\delta)$;*

   (b) *for $j \neq j_1 \in \tau(\mathcal{Y})$, assuming $j = \tau(g_1(y))$, then $\Lambda_{5,j,r_{g_1\cdot y}}^{(t)} = \frac{1}{3}B \pm O(1)$ and $\Lambda_{5,j,r}^{(t)} \ll -\varrho$ for $r \in \widehat{\mathfrak{A}}_j \setminus \{r_{g_1\cdot y}\}$.*

2. *$\ell = 2$, if $\mathbf{Z}^{2,\ell-1} \notin \mathcal{E}_2$, then*

   (a) *if $g_1 = g_2 \wedge y_0 \neq y_1$,*

   i. *for $j = j_2$, $\Lambda_{5,j_2,r_{g_2\cdot y_1}}^{(t)} = \frac{1}{2}B \pm O(1)$ and $\Lambda_{5,j_2,r}^{(t)} \ll -\varrho$ for $r \in \widehat{\mathfrak{A}}_{j_2} \setminus \{r_{g_2\cdot y_1}\}$;*

ii. *for* $j = \tau(g_2(y_0))$, $\Lambda^{(t)}_{5,j_2,r_{g_2 \cdot y_0}} = \frac{1}{2}B \pm O(1)$ *and* $\Lambda^{(t)}_{5,j,r} \ll -\varrho$ *for* $r \in \widehat{\mathfrak{A}}_j \setminus \{r_{g_2 \cdot y_0}\}$;

iii. *for other* $j \in \tau(\mathcal{Y})$, *assuming* $j = \tau(g_2(y))$, $|\Lambda^{(t)}_{5,j,r_{g_2 \cdot y}}| \le O(1)$ *and* $\Lambda^{(t)}_{5,j,r} \ll -\varrho$ *for* $r \in \widehat{\mathfrak{A}}_j \setminus \{r_{g_2 \cdot y}\}$.

(b) *if* $g_1 \ne g_2 \wedge y_0 = y_1$,

i. *for* $j = j_2$, $\Lambda^{(t)}_{5,j_2,r_{g_2 \cdot y_1}} = \frac{1}{2}B \pm O(1)$ *and* $\Lambda^{(t)}_{5,j_2,r} \ll -\varrho$ *for* $r \in \widehat{\mathfrak{A}}_{j_2} \setminus \{r_{g_2 \cdot y_1}\}$;

ii. *for* $j = \tau(g_1(y_1))$, $\Lambda^{(t)}_{5,j,r_{g_1 \cdot y_1}} = \frac{1}{2}B \pm O(1)$ *and* $\Lambda^{(t)}_{5,j,r} \ll -\varrho$ *for* $r \in \widehat{\mathfrak{A}}_j \setminus \{r_{g_1 \cdot y_1}\}$;

iii. *for other* $j \in \tau(\mathcal{Y})$, *assuming* $j = \tau(g(y_1))$, $|\Lambda^{(t)}_{5,j,r_{g \cdot y_1}}| \le O(1)$ *and* $\Lambda^{(t)}_{5,j,r} \ll -\varrho$ *for* $r \in \widehat{\mathfrak{A}}_j \setminus \{r_{g \cdot y_1}\}$.

(c) *if* $g_1 = g_2 \wedge y_0 = y_1$,

i. *for* $j = j_2$, $\widehat{\mathfrak{A}}_{j_2} = \{r_{g_2 \cdot y_1}\}$, $\Lambda^{(t)}_{5,j_2,r_{g_2 \cdot y_1}} = B \pm O(\delta)$ ;

ii. *for* $j \ne j_2 \in \tau(\mathcal{Y})$, $|\Lambda^{(t)}_{5,j,r}| \le O(1)$ *for all* $r \in \widehat{\mathfrak{A}}_j$.

(d) $g_1 \ne g_2 \wedge y_0 \ne y_1 \wedge \left(g_1(y_0) = g_2(y_1) \vee g_2(y_0) = g_1(y_1)\right)$, $|\Lambda^{(t)}_{5,j,r}| \le O(1)$ *for all* $r \in \widehat{\mathfrak{A}}_j$.

With the characterization of activated neurons, we can derive the following logits for the non-high probability event.

**Lemma G.15.** *If Induction G.1 holds for all iterations* $< t$, *given input* $\mathbf{Z}^{2,\ell-1}$, *then we have*

1. *for* $\ell = 1$, *if* $\mathbf{Z}^{2,\ell-1} \notin \mathcal{E}_1$, *then* $1 - \mathbf{logit}^{(t)}_{5,j_1} = O\left(\frac{1}{\text{poly}d}\right)$.

2. $\ell = 2$, *if* $\mathbf{Z}^{2,\ell-1} \notin \mathcal{E}_2$, *then*

   (a) *if* $g_1 = g_2 \wedge y_0 \ne y_1$, $\mathbf{logit}^{(t)}_{5,j} = \Omega(1)$ *for* $j \in \{j_2, \tau(g_2(y_0))\}$, $1 - \mathbf{logit}^{(t)}_{5,j_2} - \mathbf{logit}^{(t)}_{5,\tau(g_2(y_0))} = \frac{1}{\text{poly}d}$.

   (b) *if* $g_1 \ne g_2 \wedge y_0 = y_1$, $\mathbf{logit}^{(t)}_{5,j} = \Omega(1)$ *for* $j \in \{j_2, \tau(g_1(y_1))\}$, $1 - \mathbf{logit}^{(t)}_{5,j_2} - \mathbf{logit}^{(t)}_{5,\tau(g_1(y_1))} = \frac{1}{\text{poly}d}$.

   (c) *if* $g_1 = g_2 \wedge y_0 = y_1$, $1 - \mathbf{logit}^{(t)}_{5,j_2} = O\left(\frac{1}{\text{poly}d}\right)$.

   (d) $g_1 \ne g_2 \wedge y_0 \ne y_1 \wedge \left(g_1(y_0) = g_2(y_1) \vee g_2(y_0) = g_1(y_1)\right)$, $\mathbf{logit}^{(t)}_{5,j_2} = O(\frac{1}{d})$ *for all* $j$.

### G.3.2 Gradient Lemma

**Lemma G.16.** *If Induction G.1 holds for all iterations* $< t$, *given* $s \in \tau(\mathcal{X})$, *for* $[\mathbf{Q}^{(t)}_{4,3}]_{s,s}$, *we have*

$$\sum_{\ell=1}^{2} \left[ -\nabla_{\mathbf{Q}^{(t)}_{4,3}} \mathsf{Loss}^{2,\ell}_5 \right]_{s,s} = \Theta\left(\frac{\log d}{d}\right).$$

*Proof.* By gradient decompositions, we have

$$\sum_{\ell=1}^{2} \left[ -\nabla_{\mathbf{Q}^{(t)}_{4,3}} \mathsf{Loss}^{2,\ell}_5 \right]_{s,s} = \sum_{\ell \in [2]} \sum_{\kappa \in \{i,ii,iii\}} \mathcal{N}^{(t)}_{s,3,\ell,\kappa}.$$

By Lemma G.13 and Lemma G.15, it is straightforward to see that $|\mathcal{N}^{(t)}_{s,3,1,iii}|, |\mathcal{N}^{(t)}_{s,3,2,iii}| = O(\frac{1}{\text{poly}d})$, and thus we can focus on other terms.

By Lemma G.11 and Lemma G.14, we have

$$\mathcal{N}^{(t)}_{s,3,1,i} = \mathbb{E}\left[ \mathbf{Attn}^{(t)}_{\text{ans},0 \to \text{pred},1} \cdot (1 - \mathbf{logit}^{(t)}_{5,j_1}) \cdot \right.$$

$$\left(\left(V_{j_1, r_{g_1 \cdot y_0}}(g_1) - \Lambda^{(t)}_{5,j_1,r_{g_1 \cdot y_0}} \pm \widetilde{O}(\sigma_0)\right) \pm \widetilde{O}(\delta^q)\right)\mathbb{1}_{\{\tau(x_0)=s\}\cap\mathcal{E}_1}\right]$$

$$+ \mathbb{E}\left[\mathbf{Attn}^{(t)}_{\mathsf{ans},0\to\mathsf{pred},1} \cdot (1 - \mathbf{logit}^{(t)}_{5,j_1})\cdot\right.$$

$$\left.\left(\left(V_{j_1, r_{g_1 \cdot y_0}}(g_1) - \Lambda^{(t)}_{5,j_1,r_{g_1 \cdot y_0}} \pm \widetilde{O}(\sigma_0)\right) \pm \widetilde{O}(\delta^q)\right)\mathbb{1}_{\{\tau(x_0)=s\}\cap\mathcal{E}_1^c}\right]$$

$$\overset{(a)}{=} \Theta(\frac{1}{d}) \cdot \Omega(1) \cdot \Theta(B) + \Theta(\frac{1}{d}) \cdot O(\frac{1}{\mathsf{poly}d}) \cdot \Theta(B) \cdot O(\frac{1}{\log d})$$

$$= \Theta\left(\frac{\log d}{d}\right),$$

where (a) follows from Lemma G.12, Lemma G.13, Lemma G.14 and Lemma G.15, and the fact that $\tau(x_0) = s$ holds with probability $\frac{1}{d}$. $\mathcal{N}^{(t)}_{s,3,2,i}$ can be upper bounded similarly.

Moving to $\mathcal{N}^{(t)}_{s,3,1,ii}$, noticing that $V_{j_1', r_{g_2 \cdot y_0}}(g_1) = -B + O(\delta)$ on $\mathcal{E}_1$, we have

$$\mathcal{N}^{(t)}_{s,3,1,ii} = -\mathbb{E}\left[\mathbf{Attn}^{(t)}_{\mathsf{ans},0\to\mathsf{pred},1} \cdot \mathbf{logit}^{(t)}_{5,j_1'}\cdot\right.$$

$$\left.\left(\left(V_{j_1', r_{g_2 \cdot y_0}}(g_1) - \Lambda^{(t)}_{5,j_1',r_{g_2 \cdot y_0}} \pm \widetilde{O}(\sigma_0)\right) \pm \widetilde{O}(\delta^q)\right)\mathbb{1}_{\{\tau(x_0)=s\}\cap\mathcal{E}_1}\right]$$

$$- \mathbb{E}\left[\mathbf{Attn}^{(t)}_{\mathsf{ans},0\to\mathsf{pred},1} \cdot \sum_{y\neq y_0\in\mathcal{Y}} \mathbf{logit}^{(t)}_{5,\tau(g_1(y))}\cdot\right.$$

$$\left.\left(\left(V_{\tau(g_1(y)), r_{g_1 \cdot y}}(g_1) - \Lambda^{(t)}_{5,\tau(g_1(y)),r_{g_1 \cdot y}} \pm \widetilde{O}(\sigma_0)\right) \pm \widetilde{O}(\delta^q)\right)\mathbb{1}_{\{\tau(x_0)=s\}\cap\mathcal{E}_1^c}\right]$$

$$= \Theta(\frac{1}{d}) \cdot \Omega(1) \cdot \Theta(B) - \Theta(\frac{1}{d}) \cdot O(\frac{1}{\mathsf{poly}d}) \cdot \Theta(B) \cdot O(\frac{1}{\log d})$$

$$= \Theta\left(\frac{\log d}{d}\right).$$

For $\mathcal{N}^{(t)}_{s,3,2,ii}$, we only need to control the negative gradient, since the positive part can be easily upper bounded by $O\left(\frac{\log d}{d}\right)$.

$$\mathcal{N}^{(t)}_{s,3,2,ii} \geq -\Theta(\frac{1}{d}) \cdot O(\frac{1}{d}) \cdot \Theta(B) - \mathbb{E}\left[\mathbf{Attn}^{(t)}_{\mathsf{ans},1\to\mathsf{pred},2} \cdot \sum_{y\neq y_1\in\mathcal{Y}} \mathbf{logit}^{(t)}_{5,\tau(g_2(y))}\cdot\right.$$

$$\left.\left(\left(V_{\tau(g_2(y)), r_{g_2 \cdot y}}(g_2) - \Lambda_{5,j,r} \pm \widetilde{O}(\sigma_0)\right) \pm \widetilde{O}(\delta^q)\right)\mathbb{1}_{\{\tau(x_0)=s,g_1=g_2,y_0\neq y_1\}}\right]$$

$$\overset{(a)}{\geq} -\Theta(\frac{1}{d}) \cdot O(\frac{1}{d}) \cdot \Theta(B) - \Theta(\frac{1}{d}) \cdot O(1) \cdot \Theta(B) \cdot O(\frac{1}{\log d})$$

$$\geq -O\left(\frac{1}{d}\right),$$

where (a) is due to the fact that $\mathbf{logit}^{(t)}_{5,\tau(g_2(y_0))} = \Omega(1)$ and $\mathbf{logit}^{(t)}_{5,\tau(g_2(y))} = O(\frac{1}{d})$ for other $y \neq y_1, y_0$. Putting everything together, we complete the proof. $\qquad\square$

**Lemma G.17** (Negative gradient). *If Induction G.1 holds for all iterations $< t$, given $s \in \tau(\mathcal{X})$, we have*

$$\sum_{\ell=1}^{2}\Big[-\nabla_{\mathbf{Q}_{4,4}^{(t)}}\mathsf{Loss}_5^{2,\ell}\Big]_{s,s} \geq -O\Big(\frac{1}{\log d}\Big)\sum_{\ell=1}^{2}\Big[-\nabla_{\mathbf{Q}_{4,3}^{(t)}}\mathsf{Loss}_5^{2,\ell}\Big]_{s,s}.$$

*Proof.* By gradient decompositions, we have

$$\sum_{\ell=1}^{2}\Big[-\nabla_{\mathbf{Q}_{4,4}^{(t)}}\mathsf{Loss}_5^{2,\ell}\Big]_{s,s} = \sum_{\ell\in[2]}\sum_{\kappa\in\{i,ii,iii\}}\mathcal{N}_{s,4,\ell,\kappa}^{(t)}.$$

Similarly as Lemma G.16, $|\mathcal{N}_{s,4,1,iii}^{(t)}|, |\mathcal{N}_{s,4,2,iii}^{(t)}| = O(\frac{1}{\mathrm{poly}d})$, and thus we can focus on other terms. By Lemma G.11, Lemma G.12, and Lemma G.14, we have

$$\mathcal{N}_{s,4,1,i}^{(t)} + \mathcal{N}_{s,4,1,ii}^{(t)} = \mathbb{E}\Bigg[\mathbf{Attn}_{\mathsf{ans},0\to\mathsf{ans},0}^{(t)}\cdot(1-\mathbf{logit}_{5,j_1}^{(t)})\cdot$$

$$\Big(\Big(V_{j_1,r_{g_1\cdot y_0}}(y_0)-\Lambda_{5,j_1,r_{g_1\cdot y_0}}^{(t)}\pm\widetilde{O}(\sigma_0)\Big)\pm\widetilde{O}(\delta^q)\Big)\mathbb{1}_{\{\tau(x_0)=s\}\cap\mathcal{E}_1}\Bigg] \tag{64}$$

$$+\mathbb{E}\Bigg[\mathbf{Attn}_{\mathsf{ans},0\to\mathsf{ans},0}^{(t)}\cdot(1-\mathbf{logit}_{5,j_1}^{(t)})\cdot$$

$$\Big(\Big(V_{j_1,r_{g_1\cdot y_0}}(y_0)-\Lambda_{5,j_1,r_{g_1\cdot y_0}}^{(t)}\pm\widetilde{O}(\sigma_0)\Big)\pm\widetilde{O}(\delta^q)\Big)\mathbb{1}_{\{\tau(x_0)=s\}\cap\mathcal{E}_1^c}\Bigg] \tag{65}$$

$$-\mathbb{E}\Bigg[\mathbf{Attn}_{\mathsf{ans},0\to\mathsf{ans},0}^{(t)}\cdot\mathbf{logit}_{5,j_1'}^{(t)}\cdot$$

$$\Big(\Big(V_{j_1',r_{g_2\cdot y_0}}(y_0)-\Lambda_{5,j_1',r_{g_2\cdot y_0}}^{(t)}\pm\widetilde{O}(\sigma_0)\Big)\pm\widetilde{O}(\delta^q)\Big)\mathbb{1}_{\{\tau(x_0)=s\}\cap\mathcal{E}_1}\Bigg] \tag{66}$$

$$-\mathbb{E}\Bigg[\mathbf{Attn}_{\mathsf{ans},0\to\mathsf{ans},0}^{(t)}\cdot\sum_{g\neq g_1\in\mathcal{G}}\mathbf{logit}_{5,\tau(g(y_0))}^{(t)}\cdot$$

$$\Big(\Big(V_{\tau(g(y_0)),r_{g\cdot y_0}}(y_0)-\Lambda_{5,\tau(g(y_0)),r_{g\cdot y_0}}^{(t)}\pm\widetilde{O}(\sigma_0)\Big)\pm\widetilde{O}(\delta^q)\Big)\mathbb{1}_{\{\tau(x_0)=s\}\cap\mathcal{E}_1^c}\Bigg]. \tag{67}$$

First, considering the event $\{\tau(x_0)=s\}\cap\mathcal{E}_1$, we have

(64) + (66)

$$=\mathbb{E}\Bigg[\mathbf{Attn}_{\mathsf{ans},0\to\mathsf{ans},0}^{(t)}\cdot\Big((1-\mathbf{logit}_{5,j_1}^{(t)})\cdot\Big(V_{j_1,r_{g_1\cdot y_1}}(y_0)-\Lambda_{5,j_1,r_{g_1\cdot y_0}}^{(t)}\pm\widetilde{O}(\sigma_0)\pm\widetilde{O}(\delta^q)\Big)$$

$$-\mathbf{logit}_{5,j_1'}^{(t)}\cdot\Big(V_{j_1',r_{g_2\cdot y_0}}(y_0)-\Lambda_{5,j_1',r_{g_2\cdot y_0}}^{(t)}\pm\widetilde{O}(\sigma_0)\pm\widetilde{O}(\delta^q)\Big)\Big)\mathbb{1}_{\{\tau(x_0)=s\}\cap\mathcal{E}_1}\Bigg]$$

$$\overset{(a)}{=}\mathbb{E}\Bigg[\mathbf{Attn}_{\mathsf{ans},0\to\mathsf{ans},0}^{(t)}\cdot\Big((1-\mathbf{logit}_{5,j_1}^{(t)})\cdot\Big(V_{j_1,r_{g_1\cdot y_1}}(y_0)-\Lambda_{5,j_1,r_{g_1\cdot y_0}}^{(t)}\pm\widetilde{O}(\sigma_0)\pm\widetilde{O}(\delta^q)\Big)$$

$$-(1-\mathbf{logit}_{5,j_1}^{(t)}-\frac{1}{\mathrm{poly}d})\cdot\Big(V_{j_1',r_{g_2\cdot y_0}}(y_0)-\Lambda_{5,j_1',r_{g_2\cdot y_0}}^{(t)}\pm\widetilde{O}(\sigma_0)\pm\widetilde{O}(\delta^q)\Big)\Big)\mathbb{1}_{\{\tau(x_0)=s\}\cap\mathcal{E}_1}\Bigg]$$

$$\geq -\mathbb{E}\left[\mathbf{Attn}_{\mathsf{ans},0\to\mathsf{ans},0}^{(t)}\cdot\left((1-\mathbf{logit}_{5,j_1}^{(t)})\cdot\right.\right.$$

$$\left.\left(V_{j_1,r_{g_1\cdot y_1}}(y_0)-V_{j_1',r_{g_2\cdot y_0}}(y_0)+\Lambda_{5,j_1',r_{g_2\cdot y_0}}^{(t)}-\Lambda_{5,j_1,r_{g_1\cdot y_0}}^{(t)}\pm\widetilde{O}(\sigma_0)\pm\widetilde{O}(\delta^q)\right)\right)\mathbb{1}_{\{\tau(x_0)=s\}\cap\mathcal{E}_1}\right]$$

$$\overset{(b)}{\geq}-\mathbb{E}\left[\mathbf{Attn}_{\mathsf{ans},0\to\mathsf{ans},0}^{(t)}\cdot\left((1-\mathbf{logit}_{5,j_1}^{(t)})\cdot\left(O(\delta)+O(1)\pm\widetilde{O}(\sigma_0)\pm\widetilde{O}(\delta^q)\right)\right)\mathbb{1}_{\{\tau(x_0)=s\}\cap\mathcal{E}_1}\right]$$

$$\geq -O\Big(\frac{\mathcal{N}_{s,3,1,i}^{(t)}}{\log d}\Big),$$

where (a) follows from Lemma G.13; (b) follows from Lemma G.12 and Lemma G.3.

Notice that (65) $\geq 0$ since $V_{j_1,r_{g_1\cdot y_0}}(y_0)-\Lambda_{5,j_1,r_{g_1\cdot y_0}}^{(t)}\geq\Omega(B)$, thus we just need to consider the possible negative gradient from (67). By Lemma G.15, we have $\sum_{g\neq g_1\in\mathcal{G}}\mathbf{logit}_{5,\tau(g(y_0))}^{(t)}\leq O(\frac{1}{\mathrm{poly}d})$ on $\{\tau(x_0)=s\}\cap\mathcal{E}_1^c$, and hence (67) $\ll\mathcal{N}_{s,3,1,i}^{(t)}$.

Moving to the gradient from $\ell=2$, it is straightforward to see that $\mathcal{N}_{s,4,2,i}^{(t)}$ is non-negative, and thus we can focus on the possible negative gradient from $\mathcal{N}_{s,4,2,ii}^{(t)}$.

$$\mathcal{N}_{s,4,2,ii}^{(t)}=-\mathbb{E}\left[\mathbf{Attn}_{\mathsf{ans},1\to\mathsf{ans},1}^{(t)}\cdot\sum_{j\neq j_2\in\tau(\mathcal{Y})}\mathbf{logit}_{5,j}^{(t)}\cdot\right.$$

$$\left.\left(\sum_{r\in\widehat{\mathfrak{A}}_j}\mathbf{sReLU}'(\Lambda_{5,j,r}^{(t)})\cdot\left(V_{j,r}(y_1)-\Lambda_{5,j,r}^{(t)}\pm\widetilde{O}(\sigma_0)\right)\pm\widetilde{O}(\delta^q)\right)\mathbb{1}_{\{\tau(x_1)=s\}\cap\mathcal{E}_2}\right]$$

$$-\mathbb{E}\left[\mathbf{Attn}_{\mathsf{ans},1\to\mathsf{ans},1}^{(t)}\cdot\sum_{j\neq j_2\in\tau(\mathcal{Y})}\mathbf{logit}_{5,j}^{(t)}\cdot\right.$$

$$\left.\left(\sum_{r\in\widehat{\mathfrak{A}}_j}\mathbf{sReLU}'(\Lambda_{5,j,r}^{(t)})\cdot\left(V_{j,r}(y_1)-\Lambda_{5,j,r}^{(t)}\pm\widetilde{O}(\sigma_0)\right)\pm\widetilde{O}(\delta^q)\right)\mathbb{1}_{\{\tau(x_1)=s\}\cap\mathcal{E}_2^c}\right]$$

$$\geq -\Theta(\frac{1}{d})\cdot O(\frac{1}{d})\cdot\Theta(B)-\Theta(\frac{1}{d})\cdot O(1)\cdot\Theta(B)\cdot O(\frac{1}{\log d})\geq -O\Big(\frac{\mathcal{N}_{s,3,1,i}^{(t)}}{\log d}\Big).$$

Putting everything together, and combining with the fact that $\sum_{\ell=1}^2\left[-\nabla_{\mathbf{Q}_{4,3}^{(t)}}\mathsf{Loss}_5^{2,\ell}\right]_{s,s}=\Theta(\mathcal{N}_{s,3,1,i}^{(t)})$ from Lemma G.16, we complete the proof. $\qquad\square$

**Lemma G.18** (Growth of gap). *If Induction G.1 holds for all iterations $< t$, given $s\in\tau(\mathcal{X})$, we have*

$$\sum_{\ell=1}^2\left[-\nabla_{\mathbf{Q}_{4,3}^{(t)}}\mathsf{Loss}_5^{2,\ell}\right]_{s,s}+\sum_{\ell=1}^2\left[\nabla_{\mathbf{Q}_{4,4}^{(t)}}\mathsf{Loss}_5^{2,\ell}\right]_{s,s}\geq\Omega\Big(\sum_{\ell=1}^2\left[-\nabla_{\mathbf{Q}_{4,3}^{(t)}}\mathsf{Loss}_5^{2,\ell}\right]_{s,s}\Big).$$

*Proof.* By gradient decompositions in (50)-(61), we have

$$\sum_{\ell=1}^2\left[-\nabla_{\mathbf{Q}_{4,3}^{(t)}}\mathsf{Loss}_5^{2,\ell}\right]_{s,s}+\sum_{\ell=1}^2\left[\nabla_{\mathbf{Q}_{4,4}^{(t)}}\mathsf{Loss}_5^{2,\ell}\right]_{s,s}=\sum_{\ell\in[2]}\sum_{\kappa\in\{i,ii,iii\}}\mathcal{N}_{s,3,\ell,\kappa}^{(t)}-\mathcal{N}_{s,4,\ell,\kappa}^{(t)}.$$

Due to Lemma G.13 and Lemma G.15, $|\mathcal{N}_{s,p,\ell,iii}^{(t)}|=O(\frac{1}{\mathrm{poly}d})$ for $p\in\{3,4\}$ and $\ell\in[2]$, we can focus on the gradient difference between $\mathbf{Q}_{4,3}^{(t)}$ and $\mathbf{Q}_{4,4}^{(t)}$ contributed by other terms.

By Lemma G.10, we have

$$\mathbf{Attn}_{\mathsf{ans},0\to\mathsf{ans},0}^{(t)}\leq\mathbf{Attn}_{\mathsf{ans},0\to\mathsf{pred},1}^{(t)},\quad\mathbf{Attn}_{\mathsf{ans},1\to\mathsf{ans},1}^{(t)}\leq\mathbf{Attn}_{\mathsf{ans},1\to\mathsf{pred},2}^{(t)}.$$

Hence, for $\ell \in [2]$, we have

$$\mathcal{N}^{(t)}_{s,3,\ell,i} - \mathcal{N}^{(t)}_{s,4,\ell,i} \geq -O(\delta) \cdot O(\frac{1}{d}) \cdot \Theta(1) \geq -O(\delta/d).$$

$$\mathcal{N}^{(t)}_{s,3,1,ii} - \mathcal{N}^{(t)}_{s,4,1,ii}$$

$$= -\mathbb{E}\Bigg[ \mathbf{Attn}^{(t)}_{\mathrm{ans},0\to\mathrm{pred},1} \cdot \mathbf{logit}^{(t)}_{5,j'_1} \cdot$$

$$\left( \left( V_{j'_1, r_{g_2 \cdot y_0}}(g_1) - \Lambda^{(t)}_{5,j'_1, r_{g_2 \cdot y_0}} \pm \widetilde{O}(\sigma_0) \right) \pm \widetilde{O}(\delta^q) \right) \mathbb{1}_{\{\tau(x_0)=s\} \cap \mathcal{E}_1} \Bigg]$$

$$+ \mathbb{E}\Bigg[ \mathbf{Attn}^{(t)}_{\mathrm{ans},0\to\mathrm{ans},0} \cdot \mathbf{logit}^{(t)}_{5,j'_1} \cdot$$

$$\left( \left( V_{j'_1, r_{g_2 \cdot y_0}}(y_0) - \Lambda^{(t)}_{5,j'_1, r_{g_2 \cdot y_0}} \pm \widetilde{O}(\sigma_0) \right) \pm \widetilde{O}(\delta^q) \right) \mathbb{1}_{\{\tau(x_0)=s\} \cap \mathcal{E}_1} \Bigg]$$

$$- \mathbb{E}\Bigg[ \mathbf{Attn}^{(t)}_{\mathrm{ans},0\to\mathrm{pred},1} \cdot \sum_{y \neq y_0 \in \mathcal{Y}} \mathbf{logit}^{(t)}_{5,\tau(g_1(y))} \cdot$$

$$\left( \left( V_{\tau(g_1(y)), r_{g_1 \cdot y}}(g_1) - \Lambda^{(t)}_{5,\tau(g_1(y)), r_{g_1 \cdot y}} \pm \widetilde{O}(\sigma_0) \right) \pm \widetilde{O}(\delta^q) \right) \mathbb{1}_{\{\tau(x_0)=s\} \cap \mathcal{E}_1^c} \Bigg]$$

$$+ \mathbb{E}\Bigg[ \mathbf{Attn}^{(t)}_{\mathrm{ans},0\to\mathrm{ans},0} \cdot \sum_{y \neq y_0 \in \mathcal{Y}} \mathbf{logit}^{(t)}_{5,\tau(g_1(y))} \cdot$$

$$\left( \left( V_{\tau(g_1(y)), r_{g_1 \cdot y}}(y_0) - \Lambda^{(t)}_{5,\tau(g_1(y)), r_{g_1 \cdot y}} \pm \widetilde{O}(\sigma_0) \right) \pm \widetilde{O}(\delta^q) \right) \mathbb{1}_{\{\tau(x_0)=s\} \cap \mathcal{E}_1^c} \Bigg]$$

$$\overset{(a)}{\geq} \Theta(\frac{1}{d}) \cdot \Omega(1) \cdot \Theta(B) - \Theta(\frac{1}{d}) \cdot O(\frac{1}{\mathsf{poly}d}) \cdot O(B) \cdot O(\frac{1}{\log d}) \geq \Omega\left( \frac{\log d}{d} \right),$$

where (a) follows from Lemma G.3 that $V_{j'_1, r_{g_2 \cdot y_0}}(g_1) - \Lambda^{(t)}_{5,j'_1, r_{g_2 \cdot y_0}} \leq -\Omega(B)$, and $V_{j'_1, r_{g_2 \cdot y_0}}(y_0) - \Lambda^{(t)}_{5,j'_1, r_{g_2 \cdot y_0}} \geq \Omega(B)$.

$$\mathcal{N}^{(t)}_{s,3,2,ii} - \mathcal{N}^{(t)}_{s,4,2,ii}$$

$$= -\mathbb{E}\Bigg[ \mathbf{Attn}^{(t)}_{\mathrm{ans},1\to\mathrm{pred},2} \cdot \sum_{j \neq j_2 \in \tau(\mathcal{Y})} \mathbf{logit}^{(t)}_{5,j} \cdot$$

$$\left( \sum_{r \in \widehat{\mathfrak{A}}_j} \mathbf{sReLU}'(\Lambda^{(t)}_{5,j,r}) \cdot \left( V_{j,r}(g_2) - \Lambda^{(t)}_{5,j,r} \pm \widetilde{O}(\sigma_0) \right) \pm \widetilde{O}(\delta^q) \right) \mathbb{1}_{\{\tau(x_1)=s\} \cap \mathcal{E}_2} \Bigg]$$

$$+ \mathbb{E}\Bigg[ \mathbf{Attn}^{(t)}_{\mathrm{ans},1\to\mathrm{ans},1} \cdot \sum_{j \neq j_2 \in \tau(\mathcal{Y})} \mathbf{logit}^{(t)}_{5,j} \cdot$$

$$\left( \sum_{r \in \widehat{\mathfrak{A}}_j} \mathbf{sReLU}'(\Lambda^{(t)}_{5,j,r}) \cdot \left( V_{j,r}(y_1) - \Lambda^{(t)}_{5,j,r} \pm \widetilde{O}(\sigma_0) \right) \pm \widetilde{O}(\delta^q) \right) \mathbb{1}_{\{\tau(x_1)=s\} \cap \mathcal{E}_2} \Bigg]$$

$$- \mathbb{E}\Bigg[ \mathbf{Attn}^{(t)}_{\mathrm{ans},1\to\mathrm{pred},2} \cdot \sum_{j \neq j_2 \in \tau(\mathcal{Y})} \mathbf{logit}^{(t)}_{5,j} \cdot$$

$$\left(\sum_{r\in\widehat{\mathfrak{A}}_j}\mathbf{sReLU}'(\Lambda_{5,j,r}^{(t)})\cdot\left(V_{j,r}(g_2)-\Lambda_{5,j,r}^{(t)}\pm\widetilde{O}(\sigma_0)\right)\pm\widetilde{O}(\delta^q)\right)\mathbb{1}_{\{\tau(x_1)=s\}\cap\mathcal{E}_2^c}\right]$$

$$+\mathbb{E}\left[\mathbf{Attn}_{\mathsf{ans},1\to\mathsf{ans},1}^{(t)}\cdot\sum_{j\neq j_2\in\tau(\mathcal{Y})}\mathbf{logit}_{5,j}^{(t)}\cdot\right.$$

$$\left.\left(\sum_{r\in\widehat{\mathfrak{A}}_j}\mathbf{sReLU}'(\Lambda_{5,j,r}^{(t)})\cdot\left(V_{j,r}(y_1)-\Lambda_{5,j,r}^{(t)}\pm\widetilde{O}(\sigma_0)\right)\pm\widetilde{O}(\delta^q)\right)\mathbb{1}_{\{\tau(x_1)=s\}\cap\mathcal{E}_2^c}\right]$$

$$\overset{(a)}{\geq}-\Theta\Big(\frac{1}{d}\Big)\cdot O\Big(\frac{1}{d}\Big)\cdot\Theta(B)-\Theta\Big(\frac{1}{d}\Big)\cdot O(1)\cdot\Theta(B)\cdot O\Big(\frac{1}{\log d}\Big)\geq-O\Big(\frac{1}{d}\Big),$$

where (a) follows from Lemma G.11 and Lemma G.13, which together imply that on the event $\mathcal{E}_2$, only a constant number of neurons are activated and $\mathbf{logit}_{5,j}^{(t)}\leq O\big(\frac{1}{d}\big)$ for all $j\neq j_2$; and from Lemma G.15, which implies that on the complement event $\mathcal{E}_2^c$, occurring with probability at most $O\left(\frac{1}{\log d}\right)$, there exists at most one $j\neq j_2$ such that $\mathbf{logit}_{5,j}^{(t)}\geq\Omega(1)$ while $\mathbf{logit}_{5,j}^{(t)}\leq O\left(\frac{1}{\mathrm{poly}d}\right)$ for other $j\in\tau(\mathcal{Y})$.

Putting it all together, we finish the proof. $\qquad\square$

**Lemma G.19.** *If Induction G.1 holds for all iterations $<t$, given $s\in\tau(\mathcal{X})$, for $[\mathbf{Q}_{4,3}^{(t)}]_{s,s}\geq\Omega(\frac{\varrho}{\log d})$, we have*

$$\sum_{\ell=1}^{2}\left[-\nabla_{\mathbf{Q}_{4,4}^{(t)}}\mathsf{Loss}_5^{2,\ell}\right]_{s,s}\geq\Omega\left(\sum_{\ell=1}^{2}\left[-\nabla_{\mathbf{Q}_{4,3}^{(t)}}\mathsf{Loss}_5^{2,\ell}\right]_{s,s}\right).$$

*Proof.* Notice that by Lemma G.17, when $[\mathbf{Q}_{4,3}^{(t)}]_{s,s}\geq\Omega(\frac{\varrho}{\log d})$, we have $[\mathbf{Q}_{4,4}^{(t)}]_{s,s}\geq-O(\frac{\varrho}{\log^2 d})$. Hence

$$\mathbf{Attn}_{\mathsf{ans},1\to\mathsf{pred},2}^{(t)}+\mathbf{Attn}_{\mathsf{ans},1\to\mathsf{ans},1}^{(t)}-\mathbf{Attn}_{\mathsf{ans},1\to\mathsf{pred},1}^{(t)}-\mathbf{Attn}_{\mathsf{ans},1\to\mathsf{ans},0}^{(t)}\geq\Omega\Big(\frac{\varrho}{\log d}\Big),$$

which implies $\Lambda_{5,j_2,r_{g_2\cdot y_1}}^{(t)}(\mathbf{Z}^{2,1})\geq\Omega(\frac{\varrho}{\log d})\cdot B\pm O(\delta)\geq\varrho$ already lies in the linear regime for $\mathbf{Z}^{2,1}\in\mathcal{E}_2$. Then we have

$$\mathcal{N}_{s,4,2,i}^{(t)}\geq\mathbb{E}\left[\mathbf{Attn}_{\mathsf{ans},1\to\mathsf{ans},1}^{(t)}\cdot\left((1-\mathbf{logit}_{5,j_2}^{(t)})\cdot\right.\right.$$

$$\left.\left.\left(V_{j_2,r_{g_2\cdot y_1}}-\Lambda_{5,j_2,r_{g_2\cdot y_1}}^{(t)}\pm\widetilde{O}(\sigma_0)\right)\pm\widetilde{O}(\delta^q)\right)\mathbb{1}_{\{\tau(x_1)=s\}\cap\mathcal{E}_2}\right]$$

$$\geq\Omega(1)\cdot\Theta(B)\cdot\Theta\Big(\frac{1}{d}\Big)\geq\Omega\Big(\frac{\log d}{d}\Big).$$

Moreover, from Lemma G.17, the magnitude of negative gradient from other $\mathcal{N}$ terms can be upper bounded by $O\left(\frac{\mathcal{N}_{s,3,1,i}^{(t)}}{\log d}\right)$. Therefore, combining with the fact that $\sum_{\ell=1}^{2}\left[-\nabla_{\mathbf{Q}_{4,3}^{(t)}}\mathsf{Loss}_5^{2,\ell}\right]_{s,s}=\Theta(\mathcal{N}_{s,3,1,i}^{(t)})$ from Lemma G.16, we complete the proof. $\qquad\square$

**Lemma G.20.** *If Induction G.1 holds for all iterations $<t$, given $s\neq s'\in\tau(\mathcal{X})$, for $p\in\{3,4\}$, we have*

$$\left|\sum_{\ell=1}^{2}\left[-\nabla_{\mathbf{Q}_{4,p}^{(t)}}\mathsf{Loss}_5^{2,\ell}\right]_{s,s'}\right|\leq O\Big(\frac{\log d}{d^2}\Big)=O\Big(\frac{1}{d}\Big)\cdot\left|\sum_{\ell=1}^{2}\left[-\nabla_{\mathbf{Q}_{4,p}^{(t)}}\mathsf{Loss}_5^{2,\ell}\right]_{s,s}\right|.$$

*Proof.* The proof follows directly by combining the expressions from Lemma G.8, Lemma G.9, and Lemma G.16, along with the fact that the event $\{\tau(x_0)=s,\tau(x_1)=s'\}$ occurs with probability $O\big(\frac{1}{d^2}\big)$. $\qquad\square$

### G.3.3 At the End of Stage 2.1

Putting gradient lemmas together, we can directly prove that Induction G.1 holds for all iterations $t$ until the end of stage 2.1, where we can conclude the following:

**Lemma G.21** (End of stage 2.1). *Given $s \in \tau(\mathcal{X})$, Induction G.1 holds for all iterations $t < T_{2,1,s} = O\left(\frac{d}{\eta \log^2 d}\right)$, then at the end of stage 2.1, we have*

(a) $[\mathbf{Q}_{4,p}^{(t)}]_{s,s} = \Omega(\frac{1}{\log d})$ *for* $p \in \{3,4\}$;

(b) $[\mathbf{Q}_{4,3}^{(t)}]_{s,s} - [\mathbf{Q}_{4,4}^{(t)}]_{s,s} \geq \Omega\left(\frac{1}{\log d}\right)$;

(c) $\left|[\mathbf{Q}_{4,p}^{(t)}]_{s,s'}\right| \leq O\left(\frac{[\mathbf{Q}_{4,p}^{(t)}]_{s,s}}{d}\right)$ *for* $s' \in \tau(\mathcal{X}) \neq s$ *for* $p \in \{3,4\}$; *otherwise,* $[\mathbf{Q}_{4,p}^{(t)}]_{s,s'} = 0$.

## G.4 Stage 2.2: Continual Growth of Diagonal Entries

In Stage 2.2, the diagonal entries $[\mathbf{Q}_{4,3}]_{s,s}$ and $[\mathbf{Q}_{4,4}]_{s,s}$ continue to grow until they reach a certain threshold. The analysis in this stage parallels that of Stage 2.1, but our focus now shifts to the gradients contributed by $\ell = 2$, as the logit at $\ell = 1$ is already near-optimal and thus contributes negligibly to the growth of $\mathbf{Q}_{4,3}$ and $\mathbf{Q}_{4,4}$.

**Induction G.2.** *Given $s \in \tau(\mathcal{X})$, let $T_{2,2,s}$ denote the first time that $[\mathbf{Q}_{4,3}]_{s,s}$ reaches $0.0001$. For all iterations $T_{2,1,s} \leq t < T_{2,2,s}$, we have the following holds*

(a) $[\mathbf{Q}_{4,3}^{(t)}]_{s,s}, [\mathbf{Q}_{4,4}^{(t)}]_{s,s} \leq O(1)$ *monotonically increases;*

(b) $[\mathbf{Q}_{4,3}^{(t)}]_{s,s} - [\mathbf{Q}_{4,4}^{(t)}]_{s,s} \in \left[\Omega(\frac{1}{\log d}), O(1)\right]$;

(c) *for* $(p,q) \in \{(4,3),(4,4)\}$, $\left|[\mathbf{Q}_{p,q}^{(t)}]_{s,s'}\right| \leq O\left(\frac{[\mathbf{Q}_{p,q}^{(t)}]_{s,s}}{d}\right)$ *for* $s' \in \tau(\mathcal{X}) \neq s$; *other* $[\mathbf{Q}_{p,q}^{(t)}]_{s,s'} = 0$.

Throughout the following analysis, instead of $\mathcal{E}_2$ defined in (63), we consider a renewed event $\widetilde{\mathcal{E}}_2$ for $\ell = 2$:

$$\widetilde{\mathcal{E}}_2 \triangleq \left\{ g_1 \neq g_2 \land y_0 \neq y_1 \right\}. \tag{68}$$

### G.4.1 Attention and Logit Preliminaries

The proof in this part proceeds analogously to the arguments in Appendix G.3.1, with the induction hypothesis from Induction G.2 incorporated. Hence, we omit the details here.

**Lemma G.22.** *If Induction G.2 holds for all iterations $\in [T_{2,1,s}, t)$, given input $\mathbf{Z}^{2,\ell-1}$, then we have*

1. *for* $\ell = 1$,

   (a) $\mathbf{Attn}_{\mathsf{ans},0\to\mathsf{pred},1}^{(t)} \in \left[\frac{1}{3} + \Omega\left(\frac{1}{\log d}\right), \frac{1}{3} + c_1\right]$, *where $c_1 > 0$ is a small constant;*

   (b) $\mathbf{Attn}_{\mathsf{ans},0\to\mathsf{pred},2}^{(t)} \in \left[\frac{1}{3} - c_2, \frac{1}{3} - \Omega\left(\frac{1}{\log d}\right)\right]$, *where $c_2 > 0$ is a small constant;*

   (c) $\mathbf{Attn}_{\mathsf{ans},0\to\mathsf{pred},2}^{(t)} + \Omega\left(\frac{1}{\log d}\right) \leq \mathbf{Attn}_{\mathsf{ans},0\to\mathsf{ans},0}^{(t)}$;

   (d) $\mathbf{Attn}_{\mathsf{ans},0\to\mathsf{pred},1}^{(t)} - \mathbf{Attn}_{\mathsf{ans},0\to\mathsf{ans},0}^{(t)} \in \left[\Omega\left(\frac{1}{\log d}\right), c_3\right]$.

2. *for* $\ell = 2$,

   (a) $\mathbf{Attn}_{\mathsf{ans},1\to\mathsf{pred},2}^{(t)} \in \left[\frac{1}{4} + \Omega\left(\frac{1}{\log d}\right), \frac{1}{4} + c_4\right]$, *where $c_4 > 0$ is a small constant;*

   (b) $\mathbf{Attn}_{\mathsf{ans},1\to\mathsf{pred},1}^{(t)}, \mathbf{Attn}_{\mathsf{ans},1\to\mathsf{ans},0}^{(t)} \in \left[\frac{1}{4} - c_5, \frac{1}{4} - \Omega\left(\frac{1}{\log d}\right)\right]$, *moreover,* $\left|\mathbf{Attn}_{\mathsf{ans},1\to\mathsf{ans},0}^{(t)} - \mathbf{Attn}_{\mathsf{ans},1\to\mathsf{pred},1}^{(t)}\right| \leq O\left(\frac{1}{d}\right)$, *where $c_5 > 0$ is a small constant;*

   (c) $\mathbf{Attn}_{\mathsf{ans},1\to\mathsf{pred},1}^{(t)}, \mathbf{Attn}_{\mathsf{ans},1\to\mathsf{ans},0}^{(t)} \leq \mathbf{Attn}_{\mathsf{ans},1\to\mathsf{ans},1}^{(t)} - \Omega\left(\frac{1}{\log d}\right)$;

   (d) $\mathbf{Attn}_{\mathsf{ans},1\to\mathsf{pred},2}^{(t)} - \mathbf{Attn}_{\mathsf{ans},1\to\mathsf{ans},1}^{(t)} \in \left[\Omega\left(\frac{1}{\log d}\right), c_6\right]$, *where $c_6 > 0$ is a small constant.*

*Notice that the constant $c_1 - c_6$ depends on the threshold $0.0001$ in Induction G.2. We choose the threshold $0.0001$ small enough to ensure $2C_B(c_1 + c_2) < 1 - c_6 C_B$ and $1 - 4c_5 C_B > 0$.*

**Lemma G.23.** *If Induction G.2 holds for all iterations $\in [T_{2,1,s}, t)$, given input $\mathbf{Z}^{2,\ell-1}$, then we have*

1. *for $\ell = 1$, if $\mathbf{Z}^{2,\ell-1} \in \mathcal{E}_1$, then*

    (a) *for $j = j_1$, $\Lambda_{5,j_1,r}^{(t)} \ll -\varrho$ for $r \in \widehat{\mathfrak{A}}_{j_1} \setminus \{r_{g_1 \cdot y_0}\}$;*

    (b) *for $j = j_1' \triangleq \tau(g_2(y_0))$, $\Lambda_{5,j_1',r}^{(t)} \ll -\varrho$ for $r \in \widehat{\mathfrak{A}}_{j_1'} \setminus \{r_{g_2 \cdot y_0}\}$;*

    (c) *for other $j \in \tau(\mathcal{Y})$, $r$ is not activated for all $r \in \widehat{\mathfrak{A}}_j$, i.e., $\Lambda_{5,j,r}^{(t)} \ll -\varrho$.*

2. *$\ell = 2$, if $\mathbf{Z}^{2,\ell-1} \in \widetilde{\mathcal{E}}_2$, then*

    (a) *for $j = j_2$, $\Lambda_{5,j_2,r}^{(t)} \ll -\varrho$ for $r \in \widehat{\mathfrak{A}}_{j_2} \setminus \{r_{g_2 \cdot y_1}\}$;*

    (b) *for $j = j_2' \triangleq \tau(g_2(y_0))$, $\Lambda_{5,j_2',r}^{(t)} \ll -\varrho$ for $r \in \widehat{\mathfrak{A}}_{j_2'} \setminus \{r_{g_2 \cdot y_0}\}$;*

    (c) *for other $j \in \tau(\mathcal{Y})$, $r$ is not activated for all $r \in \widehat{\mathfrak{A}}_j$, i.e., $\Lambda_{5,j,r}^{(t)} \ll -\varrho$.*

**Lemma G.24.** *If Induction G.2 holds for all iterations $\in [T_{2,1,s}, t)$, given input $\mathbf{Z}^{2,\ell}$, then we have*

1. *$\ell = 1$, for $\mathbf{Z}^{2,\ell-1} \in \mathcal{E}_1$,*

    (a) *$\Lambda_{5,j_1,r_{g_1 \cdot y_0}}^{(t)} \geq \left(\frac{1}{3} + \Omega\left(\frac{1}{\log d}\right)\right) B$;*

    (b) *$\Omega(1) \leq \Lambda_{5,j_1,r_{g_1 \cdot y_0}}^{(t)} - \Lambda_{5,j_1',r_{g_2 \cdot y_0}}^{(t)} \leq 2(c_1 + c_2) B$.*

2. *$\ell = 2$, for $\mathbf{Z}^{2,\ell-1} \in \widetilde{\mathcal{E}}_2$,*

    (a) *$\Lambda_{5,j_2,r_{g_2 \cdot y_1}}^{(t)} \in \left[\Omega(1), 4c_5 B\right]$;*

    (b) *$\Lambda_{5,j_2',r_{g_2 \cdot y_0}}^{(t)} \in \left[\Omega(1), c_6 B\right]$ and $\Lambda_{5,j_2,r_{g_2 \cdot y_1}}^{(t)} - \Lambda_{5,j_2',r_{g_2 \cdot y_0}}^{(t)} \geq \Omega(1)$.*

**Lemma G.25.** *If Induction G.2 holds for all iterations $\in [T_{2,1,s}, t)$, given input $\mathbf{Z}^{2,\ell-1}$, then we have*

1. *for $\ell = 1$, if $\mathbf{Z}^{2,\ell-1} \in \mathcal{E}_1$, $\mathbf{logit}_{5,j_1'}^{(t)} \geq \Omega\left(\frac{1}{d^{2C_B(c_1+c_2)}}\right)$;*

2. *for $\ell = 2$, if $\mathbf{Z}^{2,\ell-1} \in \widetilde{\mathcal{E}}_2$, $1 - \mathbf{logit}_{5,j_2}^{(t)} = \Omega(1)$, $\mathbf{logit}_{5,j_2'}^{(t)} = O\left(\frac{1}{d^{1-c_6 C_B}}\right)$.*

*Proof.* • For $\ell = 1$, we have

$$\mathbf{logit}_{5,j_1'}^{(t)} = \frac{1}{1 + e^{\Lambda_{5,j_1,r_{g_1 \cdot y_0}}^{(t)} - \Lambda_{5,j_1',r_{g_2 \cdot y_0}}^{(t)}} + O(d) \cdot e^{-\Lambda_{5,j_1',r_{g_2 \cdot y_0}}^{(t)}}}$$

$$\overset{(a)}{\geq} \frac{1}{1 + e^{2(c_1+c_2)B} + O(d) \cdot e^{-\left(\frac{1}{3} - 2c_1 - 2c_2\right)B}} \geq \Omega\left(\frac{1}{d^{2C_B(c_1+c_2)}}\right),$$

where the inequality (a) follows from Lemma G.24 and the last inequality is due to the fact that $(c_1 + c_2)$ is some sufficiently small constant s.t., $e^{-\left(\frac{1}{3} - 2c_1 - 2c_2\right)B} = 1/\text{poly}d$.

• For $\ell = 2$, we have

$$\mathbf{logit}_{j_2}^{(t)} = \frac{1}{1 + e^{-\Lambda_{5,j_2,r_{g_2 \cdot y_1}}^{(t)} + \Lambda_{5,j_2',r_{g_2 \cdot y_0}}^{(t)}} + O(d) \cdot e^{-\Lambda_{5,j_2,r_{g_2 \cdot y_1}}^{(t)}}}$$

$$\overset{(a)}{\leq} \frac{1}{1 + O(d) \cdot e^{-4c_5 B}} = O\left(\frac{1}{d^{1-4c_5 C_B}}\right),$$

where the inequality (a) follows from Lemma G.24. Similarly, we have

$$\mathbf{logit}_{j_2'}^{(t)} = \frac{1}{1 + e^{\Lambda_{5,j_2,r_{g_2 \cdot y_1}}^{(t)} - \Lambda_{5,j_2',r_{g_2 \cdot y_0}}^{(t)}} + O(d) \cdot e^{-\Lambda_{5,j_2',r_{g_2 \cdot y_0}}^{(t)}}}$$

$$\leq \frac{1}{1 + e^{\Omega(1)} + O(d) \cdot e^{-c_6 B}} = O\left(\frac{1}{d^{1-c_6 C_B}}\right).$$

$\square$

**Lemma G.26.** *If Induction G.2 holds for all iterations* $\in [T_{2,1,s}, t)$, *given input* $\mathbf{Z}^{2,\ell-1}$, *then we have*

1. *for* $\ell = 1$, *if* $\mathbf{Z}^{2,\ell-1} \notin \mathcal{E}_1$, *then*

   (a) *for* $j = j_1$, $\widehat{\mathfrak{A}}_{j_1} = \{r_{g_1 \cdot y_0}\}$, $\Lambda^{(t)}_{5,j_1,r_{g_1 \cdot y_0}} = B \pm O(\delta)$;

   (b) *for* $j \neq j_1 \in \tau(\mathcal{Y})$, *assuming* $j = \tau(g_1(y))$, *then* $\Lambda^{(t)}_{5,j,r_{g_1 \cdot y}} \in \left[\left(\frac{1}{3} - c_2\right)B, \left(\frac{1}{3} + c_1\right)B\right]$ *and* $\Lambda^{(t)}_{5,j,r} \ll -\varrho$ *for* $r \in \widehat{\mathfrak{A}}_j \setminus \{r_{g_1 \cdot y}\}$.

2. $\ell = 2$, *if* $\mathbf{Z}^{2,\ell-1} \notin \widetilde{\mathcal{E}}_2$, *then*

   (a) *if* $g_1 = g_2 \wedge y_0 \neq y_1$,

       i. *for* $j = j_2$, $\Lambda^{(t)}_{5,j_2,r_{g_2 \cdot y_1}} \in \left[\frac{1}{2}B + \Omega(1), \left(\frac{1}{2} + 2c_5\right)B\right]$ *and* $\Lambda^{(t)}_{5,j_2,r} \ll -\varrho$ *for* $r \in \widehat{\mathfrak{A}}_{j_2} \setminus \{r_{g_2 \cdot y_1}\}$;

       ii. *for* $j = \tau(g_2(y_0))$, $\Lambda^{(t)}_{5,j_2,r_{g_2 \cdot y_1}} - \Lambda^{(t)}_{5,j_2,r_{g_2 \cdot y_0}} \geq \Omega(1)$ *and* $\Lambda^{(t)}_{5,j,r} \ll -\varrho$ *for* $r \in \widehat{\mathfrak{A}}_j \setminus \{r_{g_2 \cdot y_0}\}$;

       iii. *for other* $j \in \tau(\mathcal{Y})$, *assuming* $j = \tau(g_2(y))$, $\Lambda^{(t)}_{5,j,r_{g_2 \cdot y}} \in [\Omega(1), c_6 B]$ *and* $\Lambda^{(t)}_{5,j,r} \ll -\varrho$ *for* $r \in \widehat{\mathfrak{A}}_j \setminus \{r_{g_2 \cdot y}\}$.

   (b) *if* $g_1 \neq g_2 \wedge y_0 = y_1$,

       i. *for* $j = j_2$, $\Lambda^{(t)}_{5,j_2,r_{g_2 \cdot y_1}} \in \left[\frac{1}{2}B + \Omega(1), \left(\frac{1}{2} + 2c_5\right)B\right]$ *and* $\Lambda^{(t)}_{5,j_2,r} \ll -\varrho$ *for* $r \in \widehat{\mathfrak{A}}_{j_2} \setminus \{r_{g_2 \cdot y_1}\}$;

       ii. *for* $j = \tau(g_1(y_1))$, $\Lambda^{(t)}_{5,j_2,r_{g_2 \cdot y_1}} - \Lambda^{(t)}_{5,j_2,r_{g_1 \cdot y_1}} \geq \Omega(1)$ *and* $\Lambda^{(t)}_{5,j,r} \ll -\varrho$ *for* $r \in \widehat{\mathfrak{A}}_j \setminus \{r_{g_1 \cdot y_1}\}$;

       iii. *for other* $j \in \tau(\mathcal{Y})$, $r$ *is not activted for all* $r \in \widehat{\mathfrak{A}}_j$, *i.e.,* $\Lambda^{(t)}_{5,j,r} \ll -\varrho$.

   (c) *if* $g_1 = g_2 \wedge y_0 = y_1$,

       i. *for* $j = j_2$, $\widehat{\mathfrak{A}}_{j_2} = \{r_{g_2 \cdot y_1}\}$, $\Lambda^{(t)}_{5,j_2,r_{g_2 \cdot y_1}} = B \pm O(\delta)$ ;

       ii. *for other* $j \in \tau(\mathcal{Y})$, *assuming* $j = \tau(g_2(y))$, $\Lambda^{(t)}_{5,j,r_{g_2 \cdot y}} \in [\Omega(1), c_6 B]$ *and* $\Lambda^{(t)}_{5,j,r} \ll -\varrho$ *for* $r \in \widehat{\mathfrak{A}}_j \setminus \{r_{g_2 \cdot y}\}$.

**Lemma G.27.** *If Induction G.2 holds for all iterations* $\in [T_{2,1,s}, t)$, *given input* $\mathbf{Z}^{2,\ell-1}$, *then we have*

1. *for* $\ell = 1$, *if* $\mathbf{Z}^{2,\ell-1} \notin \mathcal{E}_1$, *then* $1 - \mathbf{logit}^{(t)}_{5,j_1} = O\left(\frac{1}{\mathrm{poly}d}\right)$.

2. $\ell = 2$, *if* $\mathbf{Z}^{2,\ell-1} \notin \widetilde{\mathcal{E}}_2$, *then*

   (a) *if* $g_1 = g_2 \wedge y_0 \neq y_1$, $\mathbf{logit}^{(t)}_{5,j_2} = \Omega(1)$, $1 - \mathbf{logit}^{(t)}_{5,j_2} - \mathbf{logit}^{(t)}_{5,\tau(g_2(y_0))} = \frac{1}{\mathrm{poly}d}$.

   (b) *if* $g_1 \neq g_2 \wedge y_0 = y_1$, $\mathbf{logit}^{(t)}_{5,j_2} = \Omega(1)$, $1 - \mathbf{logit}^{(t)}_{5,j_2} - \mathbf{logit}^{(t)}_{5,\tau(g_1(y_1))} = \frac{1}{\mathrm{poly}d}$.

   (c) *if* $g_1 = g_2 \wedge y_0 = y_1$, $1 - \mathbf{logit}^{(t)}_{5,j_2} = O\left(\frac{1}{\mathrm{poly}d}\right)$.

### G.4.2 Gradient Lemma

**Lemma G.28.** *If Induction G.2 holds for all iterations* $\in [T_{2,1,s}, t)$, *given* $s \in \tau(\mathcal{X})$, *for* $[\mathbf{Q}^{(t)}_{4,3}]_{s,s}$, *we have*

$$\sum_{\ell=1}^{2} \left[ -\nabla_{\mathbf{Q}^{(t)}_{4,3}} \mathsf{Loss}_5^{2,\ell} \right]_{s,s} = \Theta\left(\frac{\log d}{d}\right).$$

*Proof.* The proof is similar to Lemma G.16, but we need to shift our focus to $\left[-\nabla_{\mathbf{Q}_{4,3}^{(t)}}\mathsf{Loss}_5^{2,2}\right]_{s,s}$.
By Lemma G.23 and Lemma G.26, we have

$$
\mathcal{N}_{s,3,2,i}^{(t)} = \mathbb{E}\Bigg[\mathbf{Attn}_{\mathsf{ans},1\to\mathsf{pred},2}^{(t)}\cdot(1-\mathbf{logit}_{5,j_2}^{(t)})\cdot
$$

$$
\left(\left(V_{j_2,r_{g_2\cdot y_1}}(g_2)-\Lambda_{5,j_2,r_{g_2\cdot y_1}}^{(t)}\pm\widetilde{O}(\sigma_0)\right)\pm\widetilde{O}(\delta^q)\right)\mathbb{1}_{\{\tau(x_1)=s\}\cap\widetilde{\mathcal{E}}_2}\Bigg]
$$

$$
+\mathbb{E}\Bigg[\mathbf{Attn}_{\mathsf{ans},1\to\mathsf{pred},2}^{(t)}\cdot(1-\mathbf{logit}_{5,j_2}^{(t)})\cdot
$$

$$
\left(\left(V_{j_2,r_{g_2\cdot y_1}}(g_2)-\Lambda_{5,j_2,r_{g_2\cdot y_1}}^{(t)}\pm\widetilde{O}(\sigma_0)\right)\pm\widetilde{O}(\delta^q)\right)\mathbb{1}_{\{\tau(x_1)=s\}\cap\widetilde{\mathcal{E}}_2^c}\Bigg]
$$

$$
\overset{(a)}{=}\Theta\Big(\frac{1}{d}\Big)\cdot\Omega(1)\cdot\Theta(B)+\Theta\Big(\frac{1}{d}\Big)\cdot O(1)\cdot\Theta(B)\cdot O\Big(\frac{1}{\log d}\Big)
$$

$$
=\Theta\Big(\frac{\log d}{d}\Big),
$$

where the inequality (a) follows from Lemma G.25 and Lemma G.26.
Furthermore, for $\mathcal{N}_{s,3,2,ii}^{(t)}$, we have

$$
\left|\mathcal{N}_{s,3,2,ii}^{(t)}\right|
$$

$$
\leq\left|\mathbb{E}\Bigg[\mathbf{Attn}_{\mathsf{ans},1\to\mathsf{pred},2}^{(t)}\cdot\mathbf{logit}_{5,j_2'}^{(t)}\cdot\right.
$$

$$
\left.\left(\left(V_{j_2',r_{g_2\cdot y_0}}(g_2)-\Lambda_{5,j,r}^{(t)}\pm\widetilde{O}(\sigma_0)\right)\pm\widetilde{O}(\delta^q)\right)\mathbb{1}_{\{\tau(x_1)=s\}\cap\widetilde{\mathcal{E}}_2}\Bigg]\right|
$$

$$
+\left|\mathbb{E}\Bigg[\mathbf{Attn}_{\mathsf{ans},1\to\mathsf{pred},2}^{(t)}\cdot\sum_{j\neq j_2\in\tau(\mathcal{Y})}\mathbf{logit}_{5,j}^{(t)}\cdot\right.
$$

$$
\left.\left(\sum_{r\in\widehat{\mathfrak{A}}_j}\mathbf{sReLU}'(\Lambda_{5,j,r}^{(t)})\cdot\left(V_{j,r}(g_2)-\Lambda_{5,j,r}^{(t)}\pm\widetilde{O}(\sigma_0)\right)\pm\widetilde{O}(\delta^q)\right)\mathbb{1}_{\{\tau(x_1)=s\}\cap\widetilde{\mathcal{E}}_2^c}\Bigg]\right|
$$

$$
\leq\Theta\Big(\frac{1}{d}\Big)\cdot O\Big(\frac{1}{d^{1-c_6C_B}}\Big)\cdot\Theta(B)+\Theta\Big(\frac{1}{d}\Big)\cdot O(1)\cdot\Theta(B)\cdot O\Big(\frac{1}{\log d}\Big)\leq O\Big(\frac{1}{d}\Big).
$$

$|\mathcal{N}_{s,3,1,i}^{(t)}|$ and $|\mathcal{N}_{s,3,2,i}^{(t)}|$ can be upper bounded by $O\big(\frac{\log d}{d}\big)$ as Lemma G.16. Thus, we complete the proof. □

**Lemma G.29.** *If Induction G.2 holds for all iterations $\in[T_{2,1,s},t)$, given $s\in\tau(\mathcal{X})$, we have*

$$
\sum_{\ell=1}^{2}\left[-\nabla_{\mathbf{Q}_{4,4}^{(t)}}\mathsf{Loss}_5^{2,\ell}\right]_{s,s}=\Theta\Big(\frac{\log d}{d}\Big).
$$

*Proof.* The proof follows a similar analysis to Lemma G.19, and we thus omit the details here. □

**Lemma G.30.** *If Induction G.2 holds for all iterations $\in[T_{2,1,s},t)$, given $s\in\tau(\mathcal{X})$, we have*

$$
\sum_{t'=T_{2,1,s}}^{t}\left(\sum_{\ell=1}^{2}\left[-\nabla_{\mathbf{Q}^{(t')}4,3}\mathsf{Loss}_5^{2,\ell}\right]_{s,s}-\sum_{\ell=1}^{2}\left[-\nabla_{\mathbf{Q}_{4,4}^{(t')}}\mathsf{Loss}_5^{2,\ell}\right]_{s,s}\right)
$$

$$
\geq-O\Big(\frac{1}{\log d}\Big)\cdot\left(\sum_{t'=T_{2,1,s}}^{t}\sum_{\ell=1}^{2}\left[-\nabla_{\mathbf{Q}_{4,4}^{(t')}}\mathsf{Loss}_5^{2,\ell}\right]_{s,s}\right).
$$

*Proof.* Following the analogous analysis as Lemma G.18, we have

$$\sum_{\ell=1}^{2}\left[-\nabla_{\mathbf{Q}_{4,3}^{(t)}}\mathsf{Loss}_5^{2,\ell}\right]_{s,s} + \sum_{\ell=1}^{2}\left[\nabla_{\mathbf{Q}_{4,4}^{(t)}}\mathsf{Loss}_5^{2,\ell}\right]_{s,s} = \sum_{\ell\in[2]}\sum_{\kappa\in\{i,ii,iii\}}\mathcal{N}_{s,3,\ell,\kappa}^{(t)} - \mathcal{N}_{s,4,\ell,\kappa}^{(t)}.$$

Meanwhile $|\mathcal{N}_{s,p,\ell,iii}^{(t)}| = O(\frac{1}{\mathsf{poly}d})$ for $p \in \{3,4\}$ and $\ell \in [2]$, we can focus on the gradient difference between $\mathbf{Q}_{4,3}^{(t)}$ and $\mathbf{Q}_{4,4}^{(t)}$ contributed by other terms.

For $\ell = 1$, since $\mathbf{Attn}_{\mathsf{ans},0\to\mathsf{pred},1}^{(t)} \geq \mathbf{Attn}_{\mathsf{ans},0\to\mathsf{ans},0}^{(t)}$, and thus it is straightforward to see that $\mathcal{N}_{s,3,1,i}^{(t)} - \mathcal{N}_{s,4,1,i}^{(t)} \geq 0$. Furthermore, we have

$$\mathcal{N}_{s,3,1,ii}^{(t)} - \mathcal{N}_{s,4,1,ii}^{(t)}$$

$$= -\mathbb{E}\left[\mathbf{Attn}_{\mathsf{ans},0\to\mathsf{pred},1}^{(t)} \cdot \mathbf{logit}_{5,j_1'}^{(t)} \cdot \right.$$

$$\left. \left(\left(V_{j_1',r_{g_2\cdot y_0}}(g_1) - \Lambda_{5,j_1',r_{g_2\cdot y_0}}^{(t)} \pm \widetilde{O}(\sigma_0)\right) \pm \widetilde{O}(\delta^q)\right)\mathbb{1}_{\{\tau(x_0)=s\}\cap\mathcal{E}_1}\right]$$

$$+ \mathbb{E}\left[\mathbf{Attn}_{\mathsf{ans},0\to\mathsf{ans},0}^{(t)} \cdot \mathbf{logit}_{5,j_1'}^{(t)} \cdot \right.$$

$$\left. \left(\left(V_{j_1',r_{g_2\cdot y_0}}(y_0) - \Lambda_{5,j_1',r_{g_2\cdot y_0}}^{(t)} \pm \widetilde{O}(\sigma_0)\right) \pm \widetilde{O}(\delta^q)\right)\mathbb{1}_{\{\tau(x_0)=s\}\cap\mathcal{E}_1}\right]$$

$$- \mathbb{E}\left[\mathbf{Attn}_{\mathsf{ans},0\to\mathsf{pred},1}^{(t)} \cdot \sum_{y\neq y_0\in\mathcal{Y}}\mathbf{logit}_{5,\tau(g_1(y))}^{(t)} \cdot \right.$$

$$\left. \left(\left(V_{\tau(g_1(y)),r_{g_1\cdot y}}(g_1) - \Lambda_{5,\tau(g_1(y)),r_{g_1\cdot y}}^{(t)} \pm \widetilde{O}(\sigma_0)\right) \pm \widetilde{O}(\delta^q)\right)\mathbb{1}_{\{\tau(x_0)=s\}\cap\mathcal{E}_1^c}\right]$$

$$+ \mathbb{E}\left[\mathbf{Attn}_{\mathsf{ans},0\to\mathsf{ans},0}^{(t)} \cdot \sum_{y\neq y_0\in\mathcal{Y}}\mathbf{logit}_{5,\tau(g_1(y))}^{(t)} \cdot \right.$$

$$\left. \left(\left(V_{\tau(g_1(y)),r_{g_1\cdot y}}(y_0) - \Lambda_{5,\tau(g_1(y)),r_{g_1\cdot y}}^{(t)} \pm \widetilde{O}(\sigma_0)\right) \pm \widetilde{O}(\delta^q)\right)\mathbb{1}_{\{\tau(x_0)=s\}\cap\mathcal{E}_1^c}\right]$$

$$\overset{(a)}{\geq} \Theta(\frac{1}{d}) \cdot \Omega\left(\frac{1}{d^{2C_B(c_1+c_2)}}\right) \cdot \Theta(B) - \Theta(\frac{1}{d}) \cdot O(\frac{1}{\mathsf{poly}d}) \cdot O(B) \cdot O(\frac{1}{\log d}) \geq \Omega\left(\frac{\log d}{d^{1+2C_B(c_1+c_2)}}\right).$$

where the inequality (a) is due to Lemma G.25 and Lemma G.27.

For $\ell = 2$, since $\mathbf{Attn}_{\mathsf{ans},\mathsf{pred},2\to\mathsf{pred},1}^{(t)} \geq \mathbf{Attn}_{\mathsf{ans},1\to\mathsf{ans},1}^{(t)}$, and thus it is straightforward to see that $\mathcal{N}_{s,3,2,i}^{(t)} - \mathcal{N}_{s,4,2,i}^{(t)} \geq 0$. Moreover, we have

$$\mathcal{N}_{s,3,2,ii}^{(t)} - \mathcal{N}_{s,4,2,ii}^{(t)}$$

$$= -\mathbb{E}\left[\mathbf{Attn}_{\mathsf{ans},1\to\mathsf{pred},2}^{(t)} \cdot \mathbf{logit}_{5,j_2'}^{(t)} \cdot \right.$$

$$\left. \left(\left(V_{j_2',r_{g_2\cdot y_0}}(g_2) - \Lambda_{5,j_2',r_{g_2\cdot y_0}}^{(t)} \pm \widetilde{O}(\sigma_0)\right) \pm \widetilde{O}(\delta^q)\right)\mathbb{1}_{\{\tau(x_1)=s\}\cap\widetilde{\mathcal{E}}_2}\right] \tag{69}$$

$$+ \mathbb{E}\left[\mathbf{Attn}_{\mathsf{ans},1\to\mathsf{ans},1}^{(t)} \cdot \mathbf{logit}_{5,j_2'}^{(t)} \cdot \right.$$

$$\left(\left(V_{j_2',r_{g_2\cdot y_0}}(y_1) - \Lambda_{5,j_2',r_{g_2\cdot y_0}}^{(t)} \pm \widetilde{O}(\sigma_0)\right) \pm \widetilde{O}(\delta^q)\right)\mathbb{1}_{\{\tau(x_1)=s\}\cap\widetilde{\mathcal{E}}_2}\right] \tag{70}$$

$$-\mathbb{E}\left[\mathbf{Attn}_{\mathsf{ans},1\to\mathsf{pred},2}^{(t)} \cdot \sum_{j\neq j_2\in\tau(\mathcal{Y})} \mathbf{logit}_{5,j}^{(t)}\cdot\right.$$

$$\left.\left(\sum_{r\in\widehat{\mathfrak{A}}_j} \mathbf{sReLU}'(\Lambda_{5,j,r}^{(t)}) \cdot \left(V_{j,r}(g_2) - \Lambda_{5,j,r}^{(t)} \pm \widetilde{O}(\sigma_0)\right) \pm \widetilde{O}(\delta^q)\right)\mathbb{1}_{\{\tau(x_1)=s\}\cap\widetilde{\mathcal{E}}_2^c}\right] \tag{71}$$

$$+\mathbb{E}\left[\mathbf{Attn}_{\mathsf{ans},1\to\mathsf{ans},1}^{(t)} \cdot \sum_{j\neq j_2\in\tau(\mathcal{Y})} \mathbf{logit}_{5,j}^{(t)}\cdot\right.$$

$$\left.\left(\sum_{r\in\widehat{\mathfrak{A}}_j} \mathbf{sReLU}'(\Lambda_{5,j,r}^{(t)}) \cdot \left(V_{j,r}(y_1) - \Lambda_{5,j,r}^{(t)} \pm \widetilde{O}(\sigma_0)\right) \pm \widetilde{O}(\delta^q)\right)\mathbb{1}_{\{\tau(x_1)=s\}\cap\widetilde{\mathcal{E}}_2^c}\right]. \tag{72}$$

By Lemma G.25, we obtain that

$$(69) + (70) \geq -O\left(\frac{1}{d}\right) \cdot O\left(\frac{1}{d^{1-c_6 C_B}}\right) \cdot \Theta(B) \geq -O\left(\frac{\log d}{d^{2-c_6 C_B}}\right).$$

By Lemma G.26, for $\widetilde{\mathcal{E}}_2^c$, we only need to consider the case that $g_1 = g_2 \wedge y_0 \neq y_1$, and we have

$(71) + (72)$

$$\geq -\mathbb{E}\left[\mathbf{Attn}_{\mathsf{ans},1\to\mathsf{pred},2}^{(t)} \cdot \sum_{y\neq y_1\in\mathcal{Y}} \mathbf{logit}_{5,\tau(g_2(y))}^{(t)}\cdot\right.$$

$$\left.\left(\left(V_{\tau(g_2(y)),r_{g_2\cdot y}}(g_2) - \Lambda_{5,\tau(g_2(y)),r_{g_2\cdot y}}^{(t)} \pm \widetilde{O}(\sigma_0)\right) \pm \widetilde{O}(\delta^q)\right)\mathbb{1}_{\{\tau(x_1)=s\}\cap\{g_1=g_2\wedge y_0\neq y_1\}}\right]$$

$$+\mathbb{E}\left[\mathbf{Attn}_{\mathsf{ans},1\to\mathsf{ans},1}^{(t)} \cdot \sum_{y\neq y_1\in\mathcal{Y}} \mathbf{logit}_{5,\tau(g_2(y))}^{(t)}\cdot\right.$$

$$\left.\left(\left(V_{\tau(g_2(y)),r_{g_2\cdot y}}(y_1) - \Lambda_{5,\tau(g_2(y)),r_{g_2\cdot y}}^{(t)} \pm \widetilde{O}(\sigma_0)\right) \pm \widetilde{O}(\delta^q)\right)\mathbb{1}_{\{\tau(x_1)=s\}\cap\{g_1=g_2\wedge y_0\neq y_1\}}\right]$$

$$\geq -\Theta\left(\frac{1}{d}\right) \cdot O(1) \cdot \Theta(B) \cdot O\left(\frac{1}{\log d}\right) \geq -O\left(\frac{1}{d}\right).$$

Putting it all together, combining with the fact that $c_6$ and $(c_1 + c_2)$ are sufficiently small, we finish the proof. $\qquad\square$

**Lemma G.31** (Lower bound of gap). *If Induction G.2 holds for all iterations $\in [T_{2,1,s}, t)$, given $s \in \tau(\mathcal{X})$, we have $[\mathbf{Q}_{4,3}^{(t)}]_{s,s} - [\mathbf{Q}_{4,4}^{(t)}]_{s,s} \geq \Omega\left(\frac{1}{\log d}\right)$.*

*Proof.* Letting $\widetilde{T}$ denote the first time that $[\mathbf{Q}_{4,3}^{(t)}]_{s,s} - [\mathbf{Q}_{4,4}^{(t)}]_{s,s} \leq \frac{1}{2}\left(\mathbf{Q}_{4,3}^{(T_{2,1,s})}]_{s,s} - [\mathbf{Q}_{4,4}^{(T_{2,1,s})}]_{s,s}\right)$, which implies that

$$\sum_{t'=T_{2,1,s}}^{\widetilde{T}} \left(\sum_{\ell=1}^{2}\left[-\nabla_{\mathbf{Q}^{(t')4,3}}\mathsf{Loss}_5^{2,\ell}\right]_{s,s} - \sum_{\ell=1}^{2}\left[-\nabla_{\mathbf{Q}_{4,4}^{(t')}}\mathsf{Loss}_5^{2,\ell}\right]_{s,s}\right).$$

Hence, by Lemma G.30, we have $[\mathbf{Q}_{4,3}^{(\widetilde{T})}]_{s,s}$ and $[\mathbf{Q}_{4,3}^{(\widetilde{T})}]_{s,s}$ reaches $\Omega(1)$. Thus, we can have a refined lower bound for $(71) + (72)$ in Lemma G.30, and obtain:

$(71) + (72) \geq -O\left(\frac{1}{\mathsf{poly}d}\right)$

$$- \mathbb{E}\Bigg[ \mathbf{Attn}^{(t)}_{\mathsf{ans},1\to\mathsf{pred},2} \cdot \mathbf{logit}^{(t)}_{5,\tau(g_2(y_0))} \cdot$$

$$\left( \Big( V_{\tau(g_2(y_0)),r_{g_2\cdot y_0}}(g_2) - \Lambda^{(t)}_{5,\tau(g_2(y_0)),r_{g_2\cdot y_0}} \pm \widetilde{O}(\sigma_0) \Big) \pm \widetilde{O}(\delta^q) \right) \mathbb{1}_{\{\tau(x_1)=s\}\cap\{g_1=g_2\wedge y_0\neq y_1\}} \Bigg]$$

$$+ \mathbb{E}\Bigg[ \mathbf{Attn}^{(t)}_{\mathsf{ans},1\to\mathsf{ans},1} \cdot \mathbf{logit}^{(t)}_{5,\tau(g_2(y_0))} \cdot$$

$$\left( \Big( V_{\tau(g_2(y_0)),r_{g_2\cdot y_0}}(y_1) - \Lambda^{(t)}_{5,\tau(g_2(y_0)),r_{g_2\cdot y_0}} \pm \widetilde{O}(\sigma_0) \Big) \pm \widetilde{O}(\delta^q) \right) \mathbb{1}_{\{\tau(x_1)=s\}\cap\{g_1=g_2\wedge y_0\neq y_1\}} \Bigg]$$

$$\overset{(a)}{\geq} -O\Big(\frac{1}{\mathrm{poly}d}\Big) - \Theta\Big(\frac{1}{d}\Big)\cdot O\Big(\frac{1}{d^{\Omega(1)}}\Big)\cdot \Theta(B)\cdot O\Big(\frac{1}{\log d}\Big) \geq -O\Big(\frac{1}{d^{1+\Omega(1)}}\Big),$$

where (a) follows from the fact that on the event $\{g_1 = g_2 \wedge y_0 \neq y_1\}$, we have $\Lambda^{(t)}_{5,j_2,r_{g_2\cdot y_1}} - \Lambda^{(t)}_{5,j_2,r_{g_2\cdot y_0}} \geq \Omega(B)$ once $[\mathbf{Q}^{(t)}_{4,3}]_{s,s}$ and $[\mathbf{Q}^{(t)}_{4,4}]_{s,s}$ reach constant magnitude, and consequently, the logit satisfies $\mathbf{logit}^{(t)}_{5,\tau(g_2(y_0))} \leq O\left(\frac{1}{d^{\Omega(1)}}\right)$.

Therefore,

$$\sum_{t'=\widetilde{T}+1}^{t} \left( \sum_{\ell=1}^{2}\Big[-\nabla_{\mathbf{Q}^{(t')}4,3}\mathsf{Loss}^{2,\ell}_5\Big]_{s,s} - \sum_{\ell=1}^{2}\Big[-\nabla_{\mathbf{Q}^{(t')}_{4,4}}\mathsf{Loss}^{2,\ell}_5\Big]_{s,s} \right) \geq -O\Big(\frac{1}{\log d\cdot d^{\Omega(1)}}\Big)\cdot O(1),$$

which means that $[\mathbf{Q}^{(t)}_{4,3}]_{s,s} - [\mathbf{Q}^{(t)}_{4,4}]_{s,s} \geq \Omega\Big(\frac{1}{\log d}\Big) - O\Big(\frac{1}{\log d\cdot d^{\Omega(1)}}\Big) \geq \Omega\Big(\frac{1}{\log d}\Big).$ $\qquad\square$

**Lemma G.32.** *If Induction G.2 holds for all iterations $\in [T_{2,1,s}, t)$, given $s' \neq s \in \tau(\mathcal{X})$, for $p \in \{3,4\}$, we have*

$$\left| \sum_{\ell=1}^{2}\Big[-\nabla_{\mathbf{Q}^{(t)}_{4,p}}\mathsf{Loss}^{2,\ell}_5\Big]_{s,s'} \right| \leq O\Big(\frac{\log d}{d^2}\Big) = O\Big(\frac{1}{d}\Big)\cdot\left| \sum_{\ell=1}^{2}\Big[-\nabla_{\mathbf{Q}^{(t)}_{4,p}}\mathsf{Loss}^{2,\ell}_5\Big]_{s,s} \right|.$$

### G.4.3 At the End of Stage 2.2

Putting gradient lemmas together, we can directly prove that Induction G.2 holds for all iterations $t$ until the end of stage 2.2, where we can conclude the following:

**Lemma G.33** (End of Stage 2.2). *Given $s \in \tau(\mathcal{X})$, Induction G.2 holds for all iterations $T_{2,1,s} \leq t < T_{2,2,s} = O\Big(\frac{d}{\eta\log d}\Big)$, then at the end of stage 2.2, we have*

*(a)* $[\mathbf{Q}^{(t)}_{4,p}]_{s,s} = \Omega(1)$ *for $p \in \{3,4\}$;*

*(b)* $[\mathbf{Q}^{(t)}_{4,3}]_{s,s} - [\mathbf{Q}^{(t)}_{4,4}]_{s,s} \in \left[\Omega\big(\frac{1}{\log d}\big), O(1)\right]$;*

*(c)* $|[\mathbf{Q}^{(t)}_{4,p}]_{s,s'}| \leq O\Big(\frac{[\mathbf{Q}^{(t)}_{4,p}]_{s,s}}{d}\Big)$ *for $s' \in \tau(\mathcal{X}) \neq s$, and other $[\mathbf{Q}_{4,p}]_{s,s'} = 0$.*

### G.5 Stage 2.3: Decrease of Gap and Convergence

After rapid growth of diagonal entries in stage 2.2, we now focus on the convergence of the attention and logit matrices, and the decrease of the gap between $[\mathbf{Q}^{(t)}_{4,3}]_{s,s}$ and $[\mathbf{Q}^{(t)}_{4,4}]_{s,s}$. Recall that

$$\epsilon^{L,\ell}_{\mathsf{attn}}\big(\mathbf{Z}^{L,\ell-1}\big) = 1 - \mathbf{Attn}_{\mathsf{ans},\ell-1\to\mathsf{pred},\ell}\big(\mathbf{Z}^{L,\ell-1}\big) - \mathbf{Attn}_{\mathsf{ans},\ell-1\to\mathsf{ans},\ell-1}\big(\mathbf{Z}^{L,\ell-1}\big),$$
$$\Delta^{L,\ell}\big(\mathbf{Z}^{L,\ell-1}\big) = \mathbf{Attn}_{\mathsf{ans},\ell-1\to\mathsf{pred},\ell}\big(\mathbf{Z}^{L,\ell-1}\big) - \mathbf{Attn}_{\mathsf{ans},\ell-1\to\mathsf{ans},\ell-1}\big(\mathbf{Z}^{L,\ell-1}\big).$$

Throught stage 2.3, we will focus on the attention gap $\Delta^{L,\ell}\big(\mathbf{Z}^{L,\ell-1}\big)$ instead of the gap of the attention matrices. We abbreviate. $\epsilon^{L,\ell}_{\mathsf{attn}}\big(\mathbf{Z}^{L,\ell-1}\big)$ and $\Delta^{L,\ell}\big(\mathbf{Z}^{L,\ell-1}\big)$ as $\epsilon^{L,\ell}_{\mathsf{attn}}$ and $\Delta^{L,\ell}$ for simplicity.

**Induction G.3.** *Given $\epsilon \geq \widetilde{\Omega}(\sigma_0)$, for $s \in \tau(\mathcal{X})$, let $T_{2,3,s}$ denote the first time that $\mathbb{E}\big[\epsilon^{2,2}_{\mathsf{attn}} \mid \tau(x_1) = s\big] \leq \epsilon$. For all iterations $T_{2,2,s} \leq t < T_{2,3,s}$, we have the following holds:*

(a) $[\mathbf{Q}_{4,3}^{(t)}]_{s,s}$ and $[\mathbf{Q}_{4,4}^{(t)}]_{s,s}$ monotonically increases $\leq \widetilde{O}(1)$;

(b) $\Delta^{2,\ell} \geq 0$ for any $\mathbf{Z}^{2,\ell}$ with $\ell \in \{1,2\}$;

(c) $\mathbf{Attn}_{\mathsf{ans},1\to\mathsf{pred},2}^{(t)} \leq 0.5 + \widetilde{c}_1$ for some small constant $\widetilde{c}_1 > 0$;

(d) for $(p,q) \in \{(4,3),(4,4)\}$, $\left|[\mathbf{Q}_{p,q}^{(t)}]_{s,s'}\right| \leq O\left(\frac{[\mathbf{Q}_{p,q}^{(t)}]_{s,s}}{d}\right)$ for $s' \in \tau(\mathcal{X}) \neq s$; other $[\mathbf{Q}_{p,q}^{(t)}]_{s,s'} = 0$.

### G.5.1 Attention and Logit Preliminaries

**Lemma G.34.** *If Induction G.3 holds for all iterations* $\in [T_{2,2,s}, t)$, *given* $\mathbf{Z}^{2,\ell-1}$ *then we have*

1. *for* $\ell = 1$,

   (a) $\mathbf{Attn}_{\mathsf{ans},0\to\mathsf{ans},0}^{(t)} \geq \Omega(1)$, *and* $\mathbf{Attn}_{\mathsf{ans},0\to\mathsf{ans},0}^{(t)} > \mathbf{Attn}_{\mathsf{ans},0\to\mathsf{pred},2}^{(t)}$;

2. *for* $\ell = 2$,

   (a) $\mathbf{Attn}_{\mathsf{ans},1\to\mathsf{ans},1}^{(t)} \geq \Omega(1)$, $\mathbf{Attn}_{\mathsf{ans},1\to\mathsf{pred},1}^{(t)}, \mathbf{Attn}_{\mathsf{ans},1\to\mathsf{ans},0}^{(t)} \leq \mathbf{Attn}_{\mathsf{ans},1\to\mathsf{ans},1}^{(t)}$;

   (b) $\left|\mathbf{Attn}_{\mathsf{ans},1\to\mathsf{ans},0}^{(t)} - \mathbf{Attn}_{\mathsf{ans},1\to\mathsf{pred},1}^{(t)}\right| \leq \widetilde{O}(\frac{1}{d})$.

*Moreover, given* $\mathbf{Z}^{2,1}$ *and corresponding* $\mathbf{Z}^{2,0}$, $\mathbf{Attn}_{\mathsf{ans},0\to\mathsf{pred},1}^{(t)}(\mathbf{Z}^{2,0}) \geq \mathbf{Attn}_{\mathsf{ans},1\to\mathsf{pred},2}^{(t)}(\mathbf{Z}^{2,1})$.

**Lemma G.35.** *If Induction G.3 holds for all iterations* $\in [T_{2,2,s}, t)$, *given input* $\mathbf{Z}^{2,\ell-1}$, *then we have*

1. *for* $\ell = 1$, *if* $\mathbf{Z}^{2,\ell-1} \in \mathcal{E}_1$, *then*

   (a) *for* $j = j_1$, $\Lambda_{5,j_1,r}^{(t)} \ll -\varrho$ *for* $r \in \widehat{\mathfrak{A}}_{j_1} \setminus \{r_{g_1 \cdot y_0}\}$;

   (b) *for* $j = j_1' \triangleq \tau\left(g_2(y_0)\right) = \tau(g_1(\widetilde{y}))$, $\Lambda_{5,j_1',r}^{(t)} \ll -\varrho$ *for* $r \in \widehat{\mathfrak{A}}_{j_1'} \setminus \{r_{g_2 \cdot y_0}, r_{g_1 \cdot \widetilde{y}}\}$;

   (c) *for other* $j \in \tau(\mathcal{Y})$, *assuming* $j = \tau(g_1(y))$ *for some* $y \neq y_0$, *then* $\Lambda_{5,j,r}^{(t)} \ll -\varrho$ *for* $r \in \widehat{\mathfrak{A}}_j \setminus \{r_{g_1 \cdot y}\}$.

2. $\ell = 2$, *if* $\mathbf{Z}^{2,\ell-1} \in \widetilde{\mathcal{E}}_2$, *then*

   (a) *for* $j = j_2$, $\Lambda_{5,j_2,r}^{(t)} \ll -\varrho$ *for* $r \in \widehat{\mathfrak{A}}_{j_2} \setminus \{r_{g_2 \cdot y_1}\}$;

   (b) *for* $j = j_2' \triangleq \tau\left(g_2(y_0)\right)$, $\Lambda_{5,j_2',r}^{(t)} \ll -\varrho$ *for* $r \in \widehat{\mathfrak{A}}_{j_2'} \setminus \{r_{g_2 \cdot y_0}\}$;

   (c) *for other* $j \in \tau(\mathcal{Y})$, *if* $j = \tau(g_2(y))$ *for some* $y \in \mathcal{Y}$, *then* $\Lambda_{5,j,r}^{(t)} \ll -\varrho$ *for* $r \in \widehat{\mathfrak{A}}_j \setminus \{r_{g_2 \cdot y}\}$.

**Lemma G.36.** *If Induction G.3 holds for all iterations* $\in [T_{2,2,s}, t)$, *given input* $\mathbf{Z}^{2,\ell-1}$, *then we have*

1. $\ell = 1$, *for* $\mathbf{Z}^{2,\ell-1} \in \mathcal{E}_1$,

   (a) $\Lambda_{5,j_1,r_{g_1 \cdot y_0}}^{(t)} = \left(1 - 2\epsilon_{\mathsf{attn}}^{2,1}\right)B \pm O(\delta) \geq \left(\frac{1}{3} + c_1\right)B$;

   (b) $\Lambda_{5,j_1,r_{g_1 \cdot y_0}}^{(t)} - \Lambda_{5,j_1',r_{g_2 \cdot y_0}}^{(t)} = 2\left(\mathbf{Attn}_{\mathsf{ans},0\to\mathsf{pred},1}^{(t)} - \mathbf{Attn}_{\mathsf{ans},0\to\mathsf{pred},2}^{(t)}\right)B \pm O(\delta) \geq 2(c_1 + c_2)B$;

   (c) *for* $y \neq y_0$, $\Lambda_{5,\tau(g_1(y)),r_{g_1 \cdot y}}^{(t)} = \left(2\mathbf{Attn}_{\mathsf{ans},0\to\mathsf{pred},1}^{(t)} - 1\right)B \pm O(\delta)$, *which is only activated if* $\mathbf{Attn}_{\mathsf{ans},0\to\mathsf{pred},1}^{(t)} > \frac{1}{2}$

2. $\ell = 2$, *for* $\mathbf{Z}^{2,\ell-1} \in \widetilde{\mathcal{E}}_2$,

   (a) $\Lambda_{5,j_2,r_{g_2 \cdot y_1}}^{(t)} = \left(1 - 2\epsilon_{\mathsf{attn}}^{2,2}\right)B \pm O(\delta) \geq 4c_5B$;

   (b) $\Lambda_{5,j_2',r_{g_2 \cdot y_0}}^{(t)} = \Delta^{2,2} \cdot B \pm O(\delta)$, *and*

   $$\Lambda_{5,j_2,r_{g_2 \cdot y_1}}^{(t)} - \Lambda_{5,j_2',r_{g_2 \cdot y_0}}^{(t)} = 2\left(\mathbf{Attn}_{\mathsf{ans},1\to\mathsf{ans},1}^{(t)} - \mathbf{Attn}_{\mathsf{ans},1\to\mathsf{ans},0}^{(t)}\right)B \pm O(\delta);$$

   (c) *for* $y \neq y_0, y_1$, $\Lambda_{5,\tau(g_2(y)),r_{g_2 \cdot y}}^{(t)} = \left(2\mathbf{Attn}_{\mathsf{ans},1\to\mathsf{pred},2}^{(t)} - 1\right)B \pm O(\delta)$, *which is only activated if* $\mathbf{Attn}_{\mathsf{ans},1\to\mathsf{pred},2}^{(t)} > \frac{1}{2}$.

**Lemma G.37.** *If Induction G.3 holds for all iterations $\in [T_{2,2,s}, t)$, given input $\mathbf{Z}^{2,\ell-1}$, then we have*

*1. for $\ell = 1$, if $\mathbf{Z}^{2,\ell-1} \in \mathcal{E}_1$,*

$$1 - \mathbf{logit}_{5,j_1}^{(t)} = \Theta(1) \cdot \mathbf{logit}_{5,j_1'}^{(t)} = \Theta\left(\frac{1}{d^{2\left(\mathbf{Attn}_{\mathsf{ans},0\to\mathsf{pred},1}^{(t)} - \mathbf{Attn}_{\mathsf{ans},0\to\mathsf{pred},2}^{(t)}\right)C_B}}\right).$$

*2. for $\ell = 2$, if $\mathbf{Z}^{2,\ell-1} \in \widetilde{\mathcal{E}}_2$,*

$$\mathbf{logit}_{5,j_2'}^{(t)} = \Theta\left(\frac{1}{d^{2\left(\mathbf{Attn}_{\mathsf{ans},1\to\mathsf{ans},1}^{(t)} - \mathbf{Attn}_{\mathsf{ans},1\to\mathsf{ans},0}^{(t)}\right)C_B} + d^{1-\Delta^{2,2}C_B}}\right),$$

*moreover,*

$$1 - \mathbf{logit}_{5,j_2}^{(t)} \geq \min\left\{\Omega(1), \Omega\left(\frac{1}{d^{C_B \cdot (1-2\epsilon_{\mathsf{attn}}^{2,2})-1}}\right)\right\}.$$

**Lemma G.38.** *If Induction G.3 holds for all iterations $\in [T_{2,2,s}, t)$, given input $\mathbf{Z}^{2,\ell-1}$, then we have*

*1. for $\ell = 1$, if $\mathbf{Z}^{2,\ell-1} \notin \mathcal{E}_1$, then*

   *(a) for $j = j_1$, $\widehat{\mathfrak{A}}_{j_1} = \{r_{g_1 \cdot y_0}\}$, $\Lambda_{5,j_1,r_{g_1 \cdot y_0}}^{(t)} = B \pm O(\delta)$;*

   *(b) for $j \neq j_1 \in \tau(\mathcal{Y})$, assuming $j = \tau(g_1(y))$ for $y \neq y_0$, then $\Lambda_{5,j_1,r_{g_1 \cdot y_0}}^{(t)} - \Lambda_{5,j,r_{g_1 \cdot y}}^{(t)} \geq 2\mathbf{Attn}_{\mathsf{ans},0\to\mathsf{ans},0}^{(t)} \cdot B \geq \Omega(B)$ and $\Lambda_{5,j,r}^{(t)} \ll -\varrho$ for $r \in \widehat{\mathfrak{A}}_j \setminus \{r_{g_1 \cdot y}\}$.*

*2. $\ell = 2$, if $\mathbf{Z}^{2,\ell-1} \notin \widetilde{\mathcal{E}}_2$, then*

   *(a) if $g_1 = g_2 \wedge y_0 \neq y_1$,*

      *i. for $j = j_2$, $\Lambda_{5,j_2,r_{g_2 \cdot y_1}}^{(t)} = (1 - \epsilon_{\mathsf{attn}}^{2,2}) \cdot B \pm O(\delta)$ and $\Lambda_{5,j_2,r}^{(t)} \ll -\varrho$ for $r \in \widehat{\mathfrak{A}}_{j_2} \setminus \{r_{g_2 \cdot y_1}\}$;*

      *ii. for $j = \tau(g_2(y_0))$, $\Lambda_{5,j_2,r_{g_2 \cdot y_1}}^{(t)} - \Lambda_{5,j,r_{g_2 \cdot y_0}}^{(t)} = 2(\mathbf{Attn}_{\mathsf{ans},1\to\mathsf{ans},1}^{(t)} - \mathbf{Attn}_{\mathsf{ans},1\to\mathsf{ans},0}^{(t)})B \pm O(\delta)$ and $\Lambda_{5,j,r}^{(t)} \ll -\varrho$ for $r \in \widehat{\mathfrak{A}}_j \setminus \{r_{g_2 \cdot y_0}\}$;*

      *iii. for other $j \in \tau(\mathcal{Y})$, assuming $j = \tau(g_2(y))$ for some $y \neq y_0, y_1$, $\Lambda_{5,j_2,r_{g_2 \cdot y_1}}^{(t)} - \Lambda_{5,j,r_{g_2 \cdot y}}^{(t)} = 2\mathbf{Attn}_{\mathsf{ans},1\to\mathsf{ans},1}^{(t)}B \pm O(\delta)$ and $\Lambda_{5,j,r}^{(t)} \ll -\varrho$ for $r \in \widehat{\mathfrak{A}}_j \setminus \{r_{g_2 \cdot y}\}$.*

   *(b) if $g_1 \neq g_2 \wedge y_0 = y_1$,*

      *i. for $j = j_2$, $\Lambda_{5,j_2,r_{g_2 \cdot y_1}}^{(t)} = (1 - \epsilon_{\mathsf{attn}}^{2,2}) \cdot B \pm O(\delta)$ and $\Lambda_{5,j_2,r}^{(t)} \ll -\varrho$ for $r \in \widehat{\mathfrak{A}}_{j_2} \setminus \{r_{g_2 \cdot y_1}\}$;*

      *ii. for $j = \tau(g_1(y_1)) = \tau(g_2(\widetilde{y}))$,*

$$\Lambda_{5,j_2,r_{g_2 \cdot y_1}}^{(t)} - \Lambda_{5,j,r_{g_1 \cdot y_1}}^{(t)} = 2(\mathbf{Attn}_{\mathsf{ans},1\to\mathsf{pred},2}^{(t)} - \mathbf{Attn}_{\mathsf{ans},1\to\mathsf{pred},1}^{(t)})B \pm O(\delta)$$
$$\Lambda_{5,j_2,r_{g_2 \cdot y_1}}^{(t)} - \Lambda_{5,j,r_{g_2 \cdot \widetilde{y}}}^{(t)} = 2(\mathbf{Attn}_{\mathsf{ans},1\to\mathsf{ans},0}^{(t)} + \mathbf{Attn}_{\mathsf{ans},1\to\mathsf{ans},1}^{(t)})B \pm O(\delta),$$

      *where $r_{g_2 \cdot \widetilde{y}}$ is only activated if $\mathbf{Attn}_{\mathsf{ans},1\to\mathsf{pred},2}^{(t)} > \frac{1}{2}$. $\Lambda_{5,j,r}^{(t)} \ll -\varrho$ for $r \in \widehat{\mathfrak{A}}_j \setminus \{r_{g_1 \cdot y_1}, r_{g_2 \cdot \widetilde{y}}\}$;*

      *iii. for other $j \in \tau(\mathcal{Y})$, assuming $j = \tau(g_2(y))$ for $y \neq y_1, \widetilde{y}$, then $r_{g_2 \cdot y}$ is only activated if $\mathbf{Attn}_{\mathsf{ans},1\to\mathsf{pred},2}^{(t)} > \frac{1}{2}$ and*

$$\Lambda_{5,j_2,r_{g_2 \cdot y_1}}^{(t)} - \Lambda_{5,j_2,r_{g_2 \cdot y}}^{(t)} = 2(\mathbf{Attn}_{\mathsf{ans},1\to\mathsf{ans},0}^{(t)} + \mathbf{Attn}_{\mathsf{ans},1\to\mathsf{ans},1}^{(t)})B \pm O(\delta).$$

$$\Lambda_{5,j,r}^{(t)} \ll -\varrho \text{ for } r \in \widehat{\mathfrak{A}}_j \setminus \{r_{g_2 \cdot y}\}.$$

   *(c) if $g_1 = g_2 \wedge y_0 = y_1$,*

      *i. for $j = j_2$, $\widehat{\mathfrak{A}}_{j_2} = \{r_{g_2 \cdot y_1}\}$, $\Lambda_{5,j_2,r_{g_2 \cdot y_1}}^{(t)} = B \pm O(\delta)$ ;*

      *ii. for other $j \in \tau(\mathcal{Y})$, assuming $j = \tau(g_2(y))$,*

$$\Lambda_{5,j_2,r_{g_2 \cdot y_1}}^{(t)} - \Lambda_{5,j_2,r_{g_2 \cdot y}}^{(t)} = 2(\mathbf{Attn}_{\mathsf{ans},1\to\mathsf{ans},0}^{(t)} + \mathbf{Attn}_{\mathsf{ans},1\to\mathsf{ans},1}^{(t)})B \pm O(\delta).$$

$$\text{and } \Lambda^{(t)}_{5,j,r} \ll -\varrho \text{ for } r \in \widehat{\mathfrak{A}}_j \setminus \{r_{g_2 \cdot y}\}.$$

**Lemma G.39.** *If Induction G.3 holds for all iterations $\in [T_{2,2,s}, t)$, given input $\mathbf{Z}^{2,\ell-1}$, then we have*

1. *for $\ell = 1$, if $\mathbf{Z}^{2,\ell-1} \notin \mathcal{E}_1$, then $1 - \mathbf{logit}^{(t)}_{5,j_1} = O\left(\dfrac{1}{d^{2\mathbf{Attn}^{(t)}_{\mathsf{ans},0\to\mathsf{ans},0}C_B}}\right) = O\left(\dfrac{1}{\mathsf{poly}d}\right).$*

2. *$\ell = 2$, if $\mathbf{Z}^{2,\ell-1} \notin \widetilde{\mathcal{E}}_2$, then*

   (a) *if $g_1 = g_2 \wedge y_0 \neq y_1$, $\mathbf{logit}^{(t)}_{5,\tau(g_2(y_0))} = O\left(\dfrac{1}{d^{2\left(\mathbf{Attn}^{(t)}_{\mathsf{ans},1\to\mathsf{ans},1}-\mathbf{Attn}^{(t)}_{\mathsf{ans},1\to\mathsf{ans},0}\right)C_B}}\right).$*

   (b) *if $g_1 \neq g_2 \wedge y_0 = y_1$, $\mathbf{logit}^{(t)}_{5,\tau(g_1(y_1))} = O\left(\dfrac{1}{d^{2\left(\mathbf{Attn}^{(t)}_{\mathsf{ans},1\to\mathsf{pred},2}-\mathbf{Attn}^{(t)}_{\mathsf{ans},1\to\mathsf{pred},1}\right)C_B}}\right).$*

   (c) *if $g_1 = g_2 \wedge y_0 = y_1$, $1 - \mathbf{logit}^{(t)}_{5,j_2} = O\left(\dfrac{1}{d^{2\left(\mathbf{Attn}^{(t)}_{\mathsf{ans},1\to\mathsf{ans},0}+\mathbf{Attn}^{(t)}_{\mathsf{ans},1\to\mathsf{ans},1}\right)C_B}}\right) = \dfrac{1}{\mathsf{poly}d}.$*

### G.5.2 Gradient Lemma

**Lemma G.40.** *If Induction G.3 holds for all iterations $\in [T_{2,2,s}, t)$, given $s \in \tau(\mathcal{X})$, for $[\mathbf{Q}_{4,4}]_{s,s}$, we have*

$$\sum_{\ell=1}^{2}\left[-\nabla_{\mathbf{Q}^{(t)}_{4,4}}\mathsf{Loss}^{2,\ell}_5\right]_{s,s} \geq \Omega\left(\frac{\epsilon \log d}{d^{(1-2\epsilon)C_B}}\right).$$

*Proof.* By gradient decomposition, we have

$$\sum_{\ell=1}^{2}\left[-\nabla_{\mathbf{Q}^{(t)}_{4,4}}\mathsf{Loss}^{2,\ell}_5\right]_{s,s} = \sum_{\ell\in[2]}\sum_{\kappa\in\{i,ii,iii\}}\mathcal{N}^{(t)}_{s,4,\ell,\kappa}.$$

Firstly, for $\mathcal{N}^{(t)}_{s,4,2,i}$, by Lemma G.35 and Lemma G.38, we have

$$\mathcal{N}^{(t)}_{s,4,2,i} = \mathbb{E}\left[\mathbf{Attn}^{(t)}_{\mathsf{ans},1\to\mathsf{ans},1}\cdot(1-\mathbf{logit}^{(t)}_{5,j_2})\cdot\right.$$

$$\left.\left(\left(V_{j_2,r_{g_2\cdot y_1}}(y_1)-\Lambda^{(t)}_{5,j_2,r_{g_2\cdot y_1}}\pm\widetilde{O}(\sigma_0)\right)\pm\widetilde{O}(\delta^q)\right)\mathbb{1}_{\{\tau(x_1)=s\}\cap\widetilde{\mathcal{E}}_2}\right]$$

$$+ \mathbb{E}\left[\mathbf{Attn}^{(t)}_{\mathsf{ans},1\to\mathsf{ans},1}\cdot(1-\mathbf{logit}^{(t)}_{5,j_2})\cdot\right.$$

$$\left.\left(\left(V_{j_2,r_{g_2\cdot y_1}}(y_1)-\Lambda^{(t)}_{5,j_2,r_{g_2\cdot y_1}}\pm\widetilde{O}(\sigma_0)\right)\pm\widetilde{O}(\delta^q)\right)\mathbb{1}_{\{\tau(x_1)=s\}\cap\widetilde{\mathcal{E}}^c_2}\right]$$

$$\overset{(a)}{\geq} \Theta\left(\frac{1}{d}\right)\cdot\mathbb{E}\left[\min\left\{\Omega(1),\Omega\left(\frac{1}{d^{C_B\cdot(1-2\epsilon^{2,2}_{\mathsf{attn}})-1}}\right)\right\}\cdot 2\epsilon^{2,2}_{\mathsf{attn}}\cdot B\mid\tau(x_1)=s\right]$$

$$\geq \Omega\left(\frac{\epsilon\log d}{d^{(1-2\epsilon)C_B}}\right), \tag{73}$$

where (a) follows from Lemma G.37. Noticing that $\mathcal{N}^{(t)}_{s,4,1,i} > 0$ and $|\mathcal{N}^{(t)}_{s,4,\ell,iii}|$ for $\ell \in [2]$ is sufficiently small, to provide a lower bound for $\sum_{\ell=1}^{2}\left[-\nabla_{\mathbf{Q}^{(t)}_{4,4}}\mathsf{Loss}^{2,\ell}_5\right]_{s,s}$, we only need to focus on the negative gradient from $\mathcal{N}^{(t)}_{s,4,1,ii}$ and $\mathcal{N}^{(t)}_{s,4,2,ii}$.

$$\mathcal{N}^{(t)}_{s,4,2,ii} =$$

$$-\mathbb{E}\left[\mathbf{Attn}^{(t)}_{\mathsf{ans},1\to\mathsf{ans},1}\cdot\mathbf{logit}^{(t)}_{5,j'_2}\cdot\right.$$

$$\left(\left(V_{j_2',r_{g_2\cdot y_0}}(y_1) - \Lambda_{5,j_2',r_{g_2\cdot y_0}}^{(t)} \pm \widetilde{O}(\sigma_0)\right) \pm \widetilde{O}(\delta^q)\right)\mathbb{1}_{\{\tau(x_1)=s\}\cap\widetilde{\mathcal{E}}_2}\Bigg]$$

$$- \mathbb{E}\Bigg[\mathbf{Attn}_{\mathsf{ans},1\to\mathsf{ans},1}^{(t)} \cdot \sum_{j\neq j_2\in\tau(\mathcal{Y})} \mathbf{logit}_{5,j}^{(t)}\cdot$$

$$\left(\sum_{r\in\widehat{\mathfrak{A}}_j} \mathbf{sReLU'}(\Lambda_{5,j,r}^{(t)}) \cdot \left(V_{j,r}(y_1) - \Lambda_{5,j,r}^{(t)} \pm \widetilde{O}(\sigma_0)\right) \pm \widetilde{O}(\delta^q)\right)\mathbb{1}_{\{\tau(x_1)=s\}\cap\widetilde{\mathcal{E}}_2^c}\Bigg]$$

$$\geq -\mathbb{E}\Bigg[\mathbf{Attn}_{\mathsf{ans},1\to\mathsf{ans},1}^{(t)} \cdot \mathbf{logit}_{5,\tau(g_1(y_1))}^{(t)} \cdot \left(\mathbf{sReLU'}(\Lambda_{5,\tau(g_1(y_1)),r_{g_1\cdot y_1}}^{(t)})\cdot\right.$$

$$\left.\left(V_{\tau(g_1(y_1)),r_{g_1\cdot y_1}}(y_1) - \Lambda_{5,\tau(g_1(y_1)),r_{g_1\cdot y_1}}^{(t)} \pm \widetilde{O}(\sigma_0)\right) \pm \widetilde{O}(\delta^q)\right)\mathbb{1}_{\{\tau(x_1)=s\}\cap\{g_1\neq g_2,y_0=y_1\}}\Bigg].$$
(74)

When $\mathbb{E}\left[C_B \cdot (1 - 2\epsilon_{\mathsf{attn}}^{2,2}) \mid \tau(x_1) = s\right] < 1$, by Lemma G.37 and Lemma G.39, we have

$$\mathbf{logit}_{5,\tau(g_1(y_1))}^{(t)}\mathbb{1}_{\{\tau(x_1)=s\}\cap\{g_1\neq g_2,y_0=y_1\}}$$

$$\leq O\left(\frac{1}{d^2\left(\mathbf{Attn}_{\mathsf{ans},1\to\mathsf{pred},2}^{(t)}-\mathbf{Attn}_{\mathsf{ans},1\to\mathsf{pred},1}^{(t)}\right)C_B}\right)\left(1 - \mathbf{logit}_{5,j_2}^{(t)}\right)\mathbb{1}_{\{\tau(x_1)=s\}\cap\widetilde{\mathcal{E}}_2}.$$

During this time, $\epsilon_{\mathsf{attn}}^{2,2} \geq \Omega(1)$, thus we can lower bound $\mathcal{N}_{s,4,2,ii}^{(t)}$ by $\mathcal{N}_{s,4,2,ii}^{(t)} \geq -O\left(\frac{1}{d^{\Omega(1)}\cdot\log d}\right)\cdot$ $\mathcal{N}_{s,4,2,i}^{(t)}$, which implies that the negative gradient from $\mathcal{N}_{s,4,2,ii}^{(t)}$ is dominated by the positive gradient from $\mathcal{N}_{s,4,2,i}^{(t)}$.

When $\mathbb{E}\left[C_B \cdot (1 - 2\epsilon_{\mathsf{attn}}^{2,2}) \mid \tau(x_1) = s\right] \geq 1$, since $2\left(\mathbf{Attn}_{\mathsf{ans},1\to\mathsf{pred},2}^{(t)} - \mathbf{Attn}_{\mathsf{ans},1\to\mathsf{pred},1}^{(t)}\right)C_B \geq (1 - 2\eta^{2,2})C_B$, we obtain

$$\mathbf{logit}_{\tau(g_1(y_1))}^{(t)}\mathbb{1}_{\{\tau(x_1)=s\}\cap\{g_1\neq g_2,y_0=y_1\}} \leq O(\frac{1}{d})\cdot(1 - \mathbf{logit}_{j_2}^{(t)})\mathbb{1}_{\{\tau(x_1)=s\}\cap\widetilde{\mathcal{E}}_2}.$$
(75)

For the event $\{g_1 \neq g_2, y_0 = y_1\}$, if $\Lambda_{5,\tau(g_1(y_1)),r_{g_1\cdot y_1}}^{(t)}$ is still in the linear regime, we have

$$\Lambda_{5,\tau(g_1(y_1)),r_{g_1\cdot y_1}}^{(t)} = \left(1 - 2\mathbf{Attn}_{\mathsf{ans},1\to\mathsf{pred},2}^{(t)}\right) \cdot B \pm O(\delta) \geq \varrho,$$

which implies

$$\epsilon_{\mathsf{attn}}^{2,2} \geq 1 - 2\mathbf{Attn}_{\mathsf{ans},1\to\mathsf{pred},2} \geq \Omega\left(\frac{\varrho}{B}\right).$$
(76)

Hence, putting (76) back to (74), and putting (75) back to (74), we can lower bound $\mathcal{N}_{s,4,2,ii}^{(t)}$ as follows $\mathcal{N}_{s,4,2,ii}^{(t)} \geq -O\left(\frac{1}{d}\right) \cdot \mathcal{N}_{s,4,2,i}^{(t)}$. If $\Lambda_{5,\tau(g_1(y_1)),r_{g_1\cdot y_1}}^{(t)}$ falls into the smoothed regime, we can upper bound $\mathbf{sReLU'}(\Lambda_{5,\tau(g_1(y_1)),r_{g_1\cdot y_1}}^{(t)})$ by $O(\frac{\epsilon_{\mathsf{attn}}^{2,2}B}{\varrho})$, and then similarly, we can obatian $\mathcal{N}_{s,4,2,ii}^{(t)} \geq -O\left(\frac{1}{d}\right) \cdot \mathcal{N}_{s,4,2,i}^{(t)}$.

Following the analogous analysis, the negative gradient from $\mathcal{N}_{s,4,1,ii}^{(t)}$ can also be dominated by the positive gradient from $\mathcal{N}_{s,4,2,i}^{(t)}$. Hence, we complete the proof. $\qquad\square$

**Lemma G.41.** *If Induction G.3 holds for all iterations* $\in [T_{2,2,s}, t)$, *given* $s \in \tau(\mathcal{X})$, *for* $[\mathbf{Q}_{4,p}]_{s,s'}$, $p \in \{3, 4\}$, $s' \neq s \in \tau(\mathcal{X})$, $\ell \in [2]$, *we have*

$$\left|\left[-\nabla_{\mathbf{Q}_{4,p}^{(t)}}\mathsf{Loss}_5^{2,\ell}\right]_{s,s'}\right| \leq O\left(\frac{1}{d}\right)\left|\left[-\nabla_{\mathbf{Q}_{4,p}^{(t)}}\mathsf{Loss}_5^{2,\ell}\right]_{s,s}\right|.$$

### G.5.3 Non-negative Gap

**Lemma G.42.** *If Induction G.3 holds for all iterations $\in [T_{2,2,s}, t)$, then at time t, we have $\Delta^{2,\ell} \geq 0$ for any $\mathbf{Z}^{2,\ell}$ with $\ell \in \{1, 2\}$.*

*Proof.* Let $\widetilde{T}$ denote the first time that $\mathbb{E}[\Delta^{2,2} \mid \tau(x_1) = s] < \alpha$, where $\alpha = \frac{\epsilon^{\frac{1}{\alpha-2}}}{\mathsf{poly}d}$.
Following the analogous analysis as Lemma G.30, we have

$$\sum_{\ell=1}^{2} \left[ -\nabla_{\mathbf{Q}_{4,3}^{(\widetilde{T})}} \mathsf{Loss}_5^{2,\ell} \right]_{s,s} + \sum_{\ell=1}^{2} \left[ \nabla_{\mathbf{Q}_{4,4}^{(\widetilde{T})}} \mathsf{Loss}_5^{2,\ell} \right]_{s,s} = \sum_{\ell \in [2]} \sum_{\kappa \in \{i, ii, iii\}} \mathcal{N}_{s,3,\ell,\kappa}^{(\widetilde{T})} - \mathcal{N}_{s,4,\ell,\kappa}^{(\widetilde{T})}.$$

We can ignore the negligible difference introduced by $\mathcal{N}_{s,p,\ell,iii}^{(\widetilde{T})}$ for $p \in \{3, 4\}$ and $\ell \in [2]$.

For $\ell = 1$, since $\mathbf{Attn}_{\mathsf{ans},0 \to \mathsf{pred},1}^{(\widetilde{T})} \geq \mathbf{Attn}_{\mathsf{ans},0 \to \mathsf{ans},0}^{(\widetilde{T})}$, and thus it is straightforward to see that $\mathcal{N}_{s,3,1,i}^{(\widetilde{T})} - \mathcal{N}_{s,4,1,i}^{(\widetilde{T})} \geq 0$. Furthermore, we have

$$\mathcal{N}_{s,3,1,ii}^{(\widetilde{T})} - \mathcal{N}_{s,4,1,ii}^{(\widetilde{T})}$$

$$= -\mathbb{E}\Bigg[ \mathbf{Attn}_{\mathsf{ans},0 \to \mathsf{pred},1}^{(\widetilde{T})} \cdot \mathbf{logit}_{5,j_1'}^{(\widetilde{T})} \cdot \mathbf{sReLU}'\big(\Lambda_{5,j_1',r_{g_2 \cdot y_0}}^{(t)}\big)$$

$$\left( \Big( V_{j_1', r_{g_2 \cdot y_0}}(g_1) - \Lambda_{5,j_1',r_{g_2 \cdot y_0}}^{(\widetilde{T})} \pm \widetilde{O}(\sigma_0) \Big) \pm \widetilde{O}(\delta^q) \right) \mathbb{1}_{\{\tau(x_0) = s\} \cap \mathcal{E}_1} \Bigg] \tag{77}$$

$$+ \mathbb{E}\Bigg[ \mathbf{Attn}_{\mathsf{ans},0 \to \mathsf{ans},0}^{(\widetilde{T})} \cdot \mathbf{logit}_{5,j_1'}^{(\widetilde{T})} \cdot \mathbf{sReLU}'\big(\Lambda_{5,j_1',r_{g_2 \cdot y_0}}^{(\widetilde{T})}\big)$$

$$\left( \Big( V_{j_1', r_{g_2 \cdot y_0}}(y_0) - \Lambda_{5,j_1',r_{g_2 \cdot y_0}}^{(\widetilde{T})} \pm \widetilde{O}(\sigma_0) \Big) \pm \widetilde{O}(\delta^q) \right) \mathbb{1}_{\{\tau(x_0) = s\} \cap \mathcal{E}_1} \Bigg] \tag{78}$$

$$- \mathbb{E}\Bigg[ \mathbf{Attn}_{\mathsf{ans},0 \to \mathsf{pred},1}^{(\widetilde{T})} \cdot \sum_{y \neq y_0 \in \mathcal{Y}} \mathbf{logit}_{5,\tau(g_1(y))}^{(\widetilde{T})} \cdot \mathbf{sReLU}'\big(\Lambda_{5,\tau(g_1(y)),r_{g_1 \cdot y}}^{(\widetilde{T})}\big)$$

$$\left( \Big( V_{\tau(g_1(y)),r_{g_1 \cdot y}}(g_1) - \Lambda_{5,\tau(g_1(y)),r_{g_1 \cdot y}}^{(\widetilde{T})} \pm \widetilde{O}(\sigma_0) \Big) \pm \widetilde{O}(\delta^q) \right) \mathbb{1}_{\{\tau(x_0) = s\} \cap \mathcal{E}_1^c} \Bigg]$$

$$+ \mathbb{E}\Bigg[ \mathbf{Attn}_{\mathsf{ans},0 \to \mathsf{ans},0}^{(\widetilde{T})} \cdot \sum_{y \neq y_0 \in \mathcal{Y}} \mathbf{logit}_{5,\tau(g_1(y))}^{(\widetilde{T})} \cdot \mathbf{sReLU}'\big(\Lambda_{5,\tau(g_1(y)),r_{g_1 \cdot y}}^{(\widetilde{T})}\big)$$

$$\left( \Big( V_{\tau(g_1(y)),r_{g_1 \cdot y}}(y_0) - \Lambda_{5,\tau(g_1(y)),r_{g_1 \cdot y}}^{(\widetilde{T})} \pm \widetilde{O}(\sigma_0) \Big) \pm \widetilde{O}(\delta^q) \right) \mathbb{1}_{\{\tau(x_0) = s\} \cap \mathcal{E}_1^c} \Bigg].$$

By Lemma G.39, we obtain that $\mathcal{N}_{s,3,1,ii}^{(\widetilde{T})} - \mathcal{N}_{s,4,1,ii}^{(\widetilde{T})}$ is dominated by (77) + (78). Moreover, by Lemma G.37, we have

$$\mathbf{logit}_{5,\tau(g_1(y))}^{(\widetilde{T})}(\mathbf{Z}^{2,0}) \geq \Omega\left( \frac{1}{d^{2\big(\mathbf{Attn}_{\mathsf{ans},0 \to \mathsf{pred},1}^{(\widetilde{T})} - \mathbf{Attn}_{\mathsf{ans},0 \to \mathsf{pred},2}^{(\widetilde{T})}\big)C_B}} \right)$$

$$\geq \Omega\left( \frac{1}{d^{2\big(\mathbf{Attn}_{\mathsf{ans},1 \to \mathsf{pred},2}^{(\widetilde{T})} - \mathbf{Attn}_{\mathsf{ans},1 \to \mathsf{pred},1}^{(\widetilde{T})}\big)C_B}} \right). \tag{79}$$

For $\ell = 2$, we have

$$\mathcal{N}_{s,3,2,i}^{(\widetilde{T})} - \mathcal{N}_{s,4,2,i}^{(\widetilde{T})}$$

$$\geq \mathbb{E}\left[ \mathbf{Attn}^{(\widetilde{T})}_{\mathsf{ans},1\to\mathsf{pred},2} \cdot (1 - \mathbf{logit}^{(\widetilde{T})}_{5,j_2}) \cdot \right.$$

$$\left. \left( \left( V_{j_2, r_{g_2 \cdot y_1}}(g_2) - \Lambda^{(\widetilde{T})}_{5,j_2,r_{g_2 \cdot y_1}} \pm \widetilde{O}(\sigma_0) \right) \pm \widetilde{O}(\delta^q) \right) \mathbb{1}_{\{\tau(x_1)=s\}\cap\widetilde{\mathcal{E}}_2} \right]$$

$$- \mathbb{E}\left[ \mathbf{Attn}^{(\widetilde{T})}_{\mathsf{ans},1\to\mathsf{ans},1} \cdot (1 - \mathbf{logit}^{(\widetilde{T})}_{5,j_2}) \cdot \right.$$

$$\left. \left( \left( V_{j_2, r_{g_2 \cdot y_1}}(y_1) - \Lambda^{(\widetilde{T})}_{5,j_2,r_{g_2 \cdot y_1}} \pm \widetilde{O}(\sigma_0) \right) \pm \widetilde{O}(\delta^q) \right) \mathbb{1}_{\{\tau(x_1)=s\}\cap\widetilde{\mathcal{E}}_2} \right]$$

$$\geq \Omega\big(\alpha\epsilon\log d\big)\mathbb{E}\left[ (1 - \mathbf{logit}^{(\widetilde{T})}_{5,j_2}) \cdot \mathbb{1}_{\tau(x_1)=s} \right], \tag{80}$$

where the last inequality follows from the definition of $\widetilde{T}$.

$$\mathcal{N}^{(\widetilde{T})}_{s,3,2,ii} - \mathcal{N}^{(\widetilde{T})}_{s,4,2,ii}$$

$$= -\mathbb{E}\left[ \mathbf{Attn}^{(\widetilde{T})}_{\mathsf{ans},1\to\mathsf{pred},2} \cdot \mathbf{logit}^{(\widetilde{T})}_{5,j_2'} \cdot \mathbf{sReLU}'\big(\Lambda^{(\widetilde{T})}_{5,j_2',r_{g_2 \cdot y_0}}\big) \right.$$

$$\left. \left( \left( V_{j_2', r_{g_2 \cdot y_0}}(g_2) - \Lambda^{(\widetilde{T})}_{5,j_2',r_{g_2 \cdot y_0}} \pm \widetilde{O}(\sigma_0) \right) \pm \widetilde{O}(\delta^q) \right) \mathbb{1}_{\{\tau(x_1)=s\}\cap\widetilde{\mathcal{E}}_2} \right] \tag{81}$$

$$+ \mathbb{E}\left[ \mathbf{Attn}^{(\widetilde{T})}_{\mathsf{ans},1\to\mathsf{ans},1} \cdot \mathbf{logit}^{(\widetilde{T})}_{5,j_2'} \cdot \mathbf{sReLU}'\big(\Lambda^{(\widetilde{T})}_{5,j_2',r_{g_2 \cdot y_0}}\big) \right.$$

$$\left. \left( \left( V_{j_2', r_{g_2 \cdot y_0}}(y_1) - \Lambda^{(\widetilde{T})}_{5,j_2',r_{g_2 \cdot y_0}} \pm \widetilde{O}(\sigma_0) \right) \pm \widetilde{O}(\delta^q) \right) \mathbb{1}_{\{\tau(x_1)=s\}\cap\widetilde{\mathcal{E}}_2} \right] \tag{82}$$

$$- \mathbb{E}\left[ \mathbf{Attn}^{(\widetilde{T})}_{\mathsf{ans},1\to\mathsf{pred},2} \cdot \sum_{j\neq j_2\in\tau(\mathcal{Y})} \mathbf{logit}^{(\widetilde{T})}_{5,j} \cdot \right.$$

$$\left. \left( \sum_{r\in\widehat{\mathfrak{A}}_j} \mathbf{sReLU}'(\Lambda^{(\widetilde{T})}_{5,j,r}) \cdot \left( V_{j,r}(g_2) - \Lambda^{(\widetilde{T})}_{5,j,r} \pm \widetilde{O}(\sigma_0) \right) \pm \widetilde{O}(\delta^q) \right) \mathbb{1}_{\{\tau(x_1)=s\}\cap\widetilde{\mathcal{E}}_2^c} \right] \tag{83}$$

$$+ \mathbb{E}\left[ \mathbf{Attn}^{(\widetilde{T})}_{\mathsf{ans},1\to\mathsf{ans},1} \cdot \sum_{j\neq j_2\in\tau(\mathcal{Y})} \mathbf{logit}^{(\widetilde{T})}_{5,j} \cdot \right.$$

$$\left. \left( \sum_{r\in\widehat{\mathfrak{A}}_j} \mathbf{sReLU}'(\Lambda^{(\widetilde{T})}_{5,j,r}) \cdot \left( V_{j,r}(y_1) - \Lambda^{(\widetilde{T})}_{5,j,r} \pm \widetilde{O}(\sigma_0) \right) \pm \widetilde{O}(\delta^q) \right) \mathbb{1}_{\{\tau(x_1)=s\}\cap\widetilde{\mathcal{E}}_2^c} \right]. \tag{84}$$

Notice that by Lemma G.36 and the definition of $\widetilde{T}$, we have

$$|(81) + (82)| \leq O\big((\alpha\log d)^{q-1}\big) \cdot \mathbb{E}\left[ \left( 1 - \mathbf{logit}^{(\widetilde{T})}_{5,j_2} \right) \cdot \mathbb{1}_{\tau(x_1)=s} \right],$$

which is dominated by (80) due to the choice of $\alpha$.

Furthermore, by Lemma G.38, for $\widetilde{\mathcal{E}}_2^c$, we only need to focus on the output $\mathbf{logit}^{(t)}_{5,\tau(g_2(y_0))}$ from the case $g_1 = g_2 \wedge y_0 \neq y_1$, and obtain

$$(83) + (84)$$

$$\geq -\mathbb{E}\left[ \mathbf{Attn}^{(\widetilde{T})}_{\mathsf{ans},1\to\mathsf{pred},2} \cdot \mathbf{logit}^{(\widetilde{T})}_{5,\tau(g_2(y_0))} \cdot \mathbf{sReLU}'\big(\Lambda^{(\widetilde{T})}_{5,\tau(g_2(y_0)),r_{g_2 \cdot y_0}}\big) \right.$$

$$\left(\left(V_{\tau(g_2(y_0)),r_{g_2\cdot y_0}}(g_2)-\Lambda^{(\widetilde{T})}_{5,\tau(g_2(y_0)),r_{g_2\cdot y_0}}\pm\widetilde{O}(\sigma_0)\right)\pm\widetilde{O}(\delta^q)\right)\mathbb{1}_{\{\tau(x_1)=s\}\cap\{g_1=g_2\wedge y_0\neq y_1\}}\right]$$

$$+\mathbb{E}\left[\mathbf{Attn}^{(\widetilde{T})}_{\mathsf{ans},1\to\mathsf{ans},1}\cdot\mathbf{logit}^{(\widetilde{T})}_{5,\tau(g_2(y_0))}\cdot\mathbf{sReLU}'\left(\Lambda^{(\widetilde{T})}_{5,\tau(g_2(y_0)),r_{g_2\cdot y_0}}\right)\right.$$

$$\left.\left(\left(V_{\tau(g_2(y_0)),r_{g_2\cdot y}}(y_1)-\Lambda^{(\widetilde{T})}_{5,\tau(g_2(y_0)),r_{g_2\cdot y_0}}\pm\widetilde{O}(\sigma_0)\right)\pm\widetilde{O}(\delta^q)\right)\mathbb{1}_{\{\tau(x_1)=s\}\cap\{g_1=g_2\wedge y_0\neq y_1\}}\right].$$

By Lemma G.39, we have

$$\mathbf{logit}^{(\widetilde{T})}_{5,\tau(g_2(y_0))}(\mathbf{Z}^{2,1})=O\left(\frac{1}{d^{2\left(\mathbf{Attn}^{(\widetilde{T})}_{\mathsf{ans},1\to\mathsf{ans},1}-\mathbf{Attn}^{(\widetilde{T})}_{\mathsf{ans},1\to\mathsf{ans},0}\right)C_B}}\right).$$

Since $\mathbb{E}[\Delta^{2,2}\mid\tau(x_1)=s]<\alpha$, where $\alpha$ is sufficiently small, then combining with (79), we have

$$\mathbf{logit}^{(\widetilde{T})}_{5,\tau(g_2(y_0))}(\mathbf{Z}^{2,1})=O(1)\cdot\mathbf{logit}^{(\widetilde{T})}_{5,\tau(g_2(y_0))}(\mathbf{Z}^{2,0}).$$

Therefore, since $\widetilde{\mathcal{E}}_2^c$ happens with probability $O(\frac{1}{\log d})$, we obtain that

$$(83)+(84)\geq-O(\frac{1}{\log d})\left(\mathcal{N}^{(t)}_{s,3,1,ii}-\mathcal{N}^{(t)}_{s,4,1,ii}\right).$$

Putting it all together, we can conclude that when $\mathbb{E}[\Delta^{2,2}\mid\tau(x_1)=s]$ reaches below $\alpha$, the gap in non-decreasing direction is guaranteed, i.e.,

$$\sum_{\ell=1}^2\left[-\nabla_{\mathbf{Q}^{(\widetilde{T})}_{4,3}}\mathsf{Loss}_5^{2,\ell}\right]_{s,s}+\sum_{\ell=1}^2\left[\nabla_{\mathbf{Q}^{(\widetilde{T})}_{4,4}}\mathsf{Loss}_5^{2,\ell}\right]_{s,s}>0,$$

which completes the proof. $\qquad\square$

### G.5.4 Upper Bound for Q

**Lemma G.43.** *If Induction G.3 holds for all iterations $\in[T_{2,2,s},t)$, given $s\in\tau(\mathcal{X})$, then at time $t$, we have $[\mathbf{Q}^{(t)}_{4,p}]_{s,s}\leq\widetilde{O}(1)$ for $p\in\{3,4\}$.*

*Proof.* Denote the first time that $[\mathbf{Q}^{(t)}_{4,3}]_{s,s}$ reaches $\Omega(\log^{1+c}d)$ for some small constant $c>0$ as $\widetilde{T}$. Then by direct calculations, we have

$$\mathbf{Attn}^{(t)}_{\mathsf{ans},0\to\mathsf{pred},2}\leq O\left(\frac{1}{e^{\Omega(\log^{1+c}d)}}\right),$$

$$\mathbf{Attn}^{(t)}_{\mathsf{ans},1\to\mathsf{pred},1}+\mathbf{Attn}^{(t)}_{\mathsf{ans},1\to\mathsf{ans},0}\leq O\left(\frac{1}{e^{\Omega(\log^{1+c}d)}}\right).$$

Moreover, by Lemma G.37, we can simply bound the logits as follows:

$$1-\mathbf{logit}^{(t)}_{j_1}\leq O\left(\frac{1}{d^{C_B\left(1-2\mathbf{Attn}^{(t)}_{\mathsf{ans},0\to\mathsf{pred},2}\right)-1}}\right)\leq O\left(\frac{1}{d^{C_B/2-1}}\right)\text{ for }\mathbf{Z}^{2,0}\in\mathcal{E}_1;$$

$$1-\mathbf{logit}^{(t)}_{j_2}\leq O\left(\frac{1}{d^{C_B\left(1-2\mathbf{Attn}^{(t)}_{\mathsf{ans},1\to\mathsf{pred},1}-2\mathbf{Attn}^{(t)}_{\mathsf{ans},1\to\mathsf{ans},0}\right)-1}}\right)$$

$$+O\left(\frac{1}{d^{2\left(\mathbf{Attn}^{(t)}_{\mathsf{ans},1\to\mathsf{ans},1}-\mathbf{Attn}^{(t)}_{\mathsf{ans},1\to\mathsf{ans},0}\right)C_B}}\right)\leq O\left(\frac{1}{d^{C_B/2-1}}\right)\text{ for }\mathbf{Z}^{2,1}\in\widetilde{\mathcal{E}}_2.$$

Thus, by focusing on $\mathcal{N}_{s,3,1,i}$ and $\mathcal{N}_{s,4,1,i}$, we have

$$\left| \sum_{\ell=1}^{2} \left[ -\nabla_{\mathbf{Q}_{4,3}^{(t)}} \mathsf{Loss}_5^{2,\ell} \right]_{s,s} \right| \leq \frac{1}{d} \cdot O\left(\frac{1}{d^{C_B/2-1}}\right) \cdot O\left(\frac{1}{e^{\Omega(\log^{1+c} d)}}\right) \cdot B \leq O\left(\frac{\log d}{e^{\Omega(\log^{1+c} d)} d^{C_B/2}}\right),$$

which implies

$$\mathbf{Q}_{4,3}^{(T)} \leq \mathbf{Q}_{4,3}^{(\widetilde{T})} + O\left(\frac{T\log d}{e^{\Omega(\log^{1+c} d)} d^{C_B/2}}\right) \leq \mathbf{Q}_{4,3}^{(\widetilde{T})} + O\left(\frac{\mathsf{poly} d \cdot \log d}{e^{\Omega(\log^{1+c} d)} d^{C_B/2}}\right) \leq O(\log^{1+c} d).$$

$\square$

### G.5.5 Attention Upper Bound

**Lemma G.44.** *If Induction G.3 holds for all iterations $\in [T_{2,2,s}, t)$, given $s \in \tau(\mathcal{X})$, then at time $t$, for any $\mathbf{Z}^{2,1}$, we have $\mathbf{Attn}_{\mathsf{ans},1\to\mathsf{pred},2}^{(t)} \leq 0.5 + \widetilde{c}_1$, where $\widetilde{c}_1 > 0$ is some small constant.*

*Proof.* Let $\widetilde{T}$ denote the first time that $\mathbb{E}[\mathbf{Attn}_{\mathsf{ans},1\to\mathsf{pred},2}^{(t)} \mid \tau(x_1) = s]$ reaches $0.5 + \widetilde{c}$. where $\widetilde{c} > 0$ is some small constant s.t., $2\widetilde{c} \cdot C_B > 1$. At $\widetilde{T}$, we have $\mathbf{Attn}_{\mathsf{ans},0\to\mathsf{pred},1}^{(\widetilde{T})} \mathbb{1}_{\tau(x_0)=s} \geq \mathbf{Attn}_{\mathsf{ans},1\to\mathsf{pred},2}^{(\widetilde{T})} \mathbb{1}_{\tau(x_1)=s}$ ; moreover, $\mathbf{Attn}_{\mathsf{ans},1\to\mathsf{ans},1}^{(\widetilde{T})}$ and $\mathbf{Attn}_{\mathsf{ans},0\to\mathsf{ans},0}^{(\widetilde{T})}$ is still at the constant level.

- For $\ell = 1$, by Lemma G.35 and Lemma G.38, for $y \in \mathcal{Y} \setminus \{y_0\}$, we have only $r_{g_1 \cdot y}$ has been activted to the linear regime for the prediction $\tau(g_1(y))$. Furthermore, we obtain

$$\sum_{y\in\mathcal{Y}\setminus\{y_0\}} \mathbf{logit}_{5,\tau(g_1(y))}^{(\widetilde{T})} \geq \frac{\log d \cdot e^{2\widetilde{c} C_B \log d}}{\log d \cdot e^{2\widetilde{c} C_B \log d} + O(d)} \left(1 - \mathbf{logit}_{5,j_1}^{(\widetilde{T})}\right)$$

$$= (1 - o(1)) \cdot \left(1 - \mathbf{logit}_{5,j_1}^{(\widetilde{T})}\right).$$

Thus,

$$\left[ -\nabla_{\mathbf{Q}_{4,3}^{(\widetilde{T})}} \mathsf{Loss}_5^{2,1} \right]_{s,s}$$

$$= \mathbb{E}\left[ \mathbf{Attn}_{\mathsf{ans},0\to\mathsf{pred},1}^{(\widetilde{T})} \cdot \left( \left(1 - \mathbf{logit}_{5,j_1}^{(\widetilde{T})}\right) \cdot \left( \left(V_{j_1, r_{g_1\cdot y_0}}(g_1) - \Lambda_{5,j_1,r_{g_1\cdot y_0}}^{(\widetilde{T})} \pm \widetilde{O}(\sigma_0)\right) \pm \widetilde{O}(\delta^q)\right) \right. \right.$$

$$- \sum_{y\in\mathcal{Y}\setminus\{y_0\}} \mathbf{logit}_{5,\tau(g_1(y))}^{(\widetilde{T})} \cdot \left( \left(V_{\tau(g_1(y)),r_{g_1\cdot y}}(g_1) - \Lambda_{5,\tau(g_1(y)),r_{g_1\cdot y}}^{(\widetilde{T})} \pm \widetilde{O}(\sigma_0)\right) \pm \widetilde{O}(\delta^q)\right)$$

$$\left. \pm \frac{1}{\mathsf{poly} d}\left(1 - \mathbf{logit}_{5,j_1}^{(\widetilde{T})}\right) \cdot \widetilde{O}(\sigma_0^q) \mathbb{1}_{\tau(x_0)=s} \right]$$

$$\leq \mathbb{E}\left[ \mathbf{Attn}_{\mathsf{ans},0\to\mathsf{pred},1}^{(\widetilde{T})} \cdot \left( \left(1 - \mathbf{logit}_{5,j_1}^{(\widetilde{T})}\right) \cdot \left( \left(2\mathbf{Attn}_{\mathsf{ans},0\to\mathsf{pred},2}^{(\widetilde{T})} \cdot B \pm \widetilde{O}(\sigma_0)\right) \pm \widetilde{O}(\delta^q)\right) \right. \right.$$

$$- \sum_{y\in\mathcal{Y}\setminus\{y_0\}} \mathbf{logit}_{5,\tau(g_1(y))}^{(\widetilde{T})} \cdot \left( \left(2\mathbf{Attn}_{\mathsf{ans},0\to\mathsf{ans},0}^{(\widetilde{T})} \cdot B \pm \widetilde{O}(\sigma_0)\right) \pm \widetilde{O}(\delta^q)\right)$$

$$\left. \pm \frac{1}{\mathsf{poly} d}\left(1 - \mathbf{logit}_{5,j_1}^{(\widetilde{T})}\right) \cdot \widetilde{O}(\sigma_0^q) \mathbb{1}_{\tau(x_0)=s} \right] < 0.$$

- For $\ell = 2$, for $\mathbf{Z}^{2,1} \in \widetilde{\mathcal{E}}_2 \cup \{g_1 = g_2, y_0 \neq y_1 x\}$, we have

$$\sum_{y\in\mathcal{Y}\setminus\{y_0,y_1\}} \mathbf{logit}_{5,\tau(g_2(y))}^{(\widetilde{T})} \geq \frac{\log d \cdot e^{2\widetilde{c} C_B \log d}}{\log d \cdot e^{2\widetilde{c} C_B \log d} + O(d)} \left(1 - \mathbf{logit}_{5,j_2}^{(\widetilde{T})} - \mathbf{logit}_{5,j_2'}^{(\widetilde{T})}\right)$$

$$= (1 - o(1)) \cdot \left(1 - \mathbf{logit}_{5,j_2}^{(\widetilde{T})} - \mathbf{logit}_{5,j_2'}^{(\widetilde{T})}\right).$$

Otherwise, we have

$$\sum_{y\in\mathcal{Y}\setminus\{y_1\}} \mathbf{logit}^{(\widetilde{T})}_{5,\tau(g_2(y))} = (1-o(1))\cdot\left(1-\mathbf{logit}^{(\widetilde{T})}_{5,j_2}-\mathbf{logit}^{(\widetilde{T})}_{5,j_2'}\right).$$

Therefore, similar to $\ell=1$, we obtain $\left[-\nabla_{\mathbf{Q}^{(\widetilde{T})}_{4,3}}\mathsf{Loss}^{2,2}_5\right]_{s,s} < 0$.

Combing the above two cases, we have $\sum_{\ell=1}^2\left[-\nabla_{\mathbf{Q}^{(\widetilde{T})}_{4,3}}\mathsf{Loss}^{2,\ell}_5\right]_{s,s} < 0$. It is also clear from Lemma G.40 that $\sum_{\ell=1}^2\left[-\nabla_{\mathbf{Q}^{(\widetilde{T})}_{4,4}}\mathsf{Loss}^{2,\ell}_5\right]_{s,s} \geq 0$. Hence $\mathbf{Attn}^{(\widetilde{T})}_{\mathsf{ans},1\to\mathsf{pred},2}$ cannot further grow once it reaches $0.5+\widetilde{c}$. $\qquad\square$

### G.5.6 Decreasing Gap at the End of Convergence

Let $\widetilde{T}$ denote the first time that $\mathbb{E}[\epsilon^{2,2}_{\mathsf{attn}}\mid\tau(x_1)=s]\leq 3\epsilon$, if $\mathbb{E}[\Delta^{2,2}\mid\tau(x_1)=s]\leq O\big(\frac{(d^{1.01}\epsilon)^{\frac{1}{q-1}}}{\log d}\big)$, then we can let $T^\star=\widetilde{T}$ and stop the training. Otherwise, we have $\mathbb{E}[\Delta^{2,2}\mid\tau(x_1)=s]\geq \Omega\big(\frac{(d^{1.01}\epsilon)^{\frac{1}{q-1}}}{\log d}\big)$. Following the similar argument as in Lemma G.40, we have the gradient contribution from $\ell=1$ is dominated by the gradient contribution from $\ell=2$. Thus, we focus on $\ell=2$, and obtain

$$\left[-\nabla_{\mathbf{Q}^{(\widetilde{T})}_{4,3}}\mathsf{Loss}^{2,2}_5\right]_{s,s}$$

$$= \mathbb{E}\Bigg[\mathbf{Attn}^{(\widetilde{T})}_{\mathsf{ans},1\to\mathsf{pred},2}\cdot\left(\left(1-\mathbf{logit}^{(\widetilde{T})}_{5,j_2}\right)\cdot\left(\left(V_{j_2,r_{g_2\cdot y_1}}(g_2)-\Lambda^{(\widetilde{T})}_{5,j_2,r_{g_2\cdot y_1}}\pm\widetilde{O}(\sigma_0)\right)\pm\widetilde{O}(\delta^q)\right)\right.$$

$$\left.\pm\frac{1}{\mathsf{poly}d}\left(1-\mathbf{logit}^{(\widetilde{T})}_{5,j_2}\right)\cdot\widetilde{O}(\sigma_0^q)\right)\mathbb{1}_{\{\tau(x_0)=s\}}\Bigg]$$

$$- \mathbb{E}\Bigg[\mathbf{Attn}^{(\widetilde{T})}_{\mathsf{ans},1\to\mathsf{pred},2}\cdot\left(\sum_{y\in\mathcal{Y}\setminus\{y_1\}}\mathbf{logit}^{(\widetilde{T})}_{5,\tau(g_2(y))}\cdot\mathbf{sReLU}'\big(\Lambda^{(\widetilde{T})}_{5,\tau(g_2(y)),r_{g_2\cdot y}}\big)\right.$$

$$\left.\cdot\left(\left(V_{\tau(g_2(y)),r_{g_2\cdot y}}(g_2)-\Lambda^{(\widetilde{T})}_{5,\tau(g_2(y)),r_{g_2\cdot y}}\pm\widetilde{O}(\sigma_0)\right)\pm\widetilde{O}(\delta^q)\right)\right)\mathbb{1}_{\{\tau(x_0)=s\}\cap\widetilde{\mathcal{E}}_2}\Bigg]$$

$$- \mathbb{E}\Bigg[\mathbf{Attn}^{(t)}_{\mathsf{ans},1\to\mathsf{pred},2}\cdot\sum_{j\neq j_2\in\tau(\mathcal{Y})}\mathbf{logit}^{(t)}_{5,j}\cdot$$

$$\left(\sum_{r\in\widehat{\mathfrak{A}}_j}\mathbf{sReLU}'(\Lambda^{(t)}_{5,j,r})\cdot\left(V_{j,r}(g_2)-\Lambda^{(t)}_{5,j,r}\pm\widetilde{O}(\sigma_0)\right)\pm\widetilde{O}(\delta^q)\right)\mathbb{1}_{\{\tau(x_1)=s\}\cap\widetilde{\mathcal{E}}_2^c}\Bigg]$$

$$\leq \mathbb{E}\left[\left(1-\mathbf{logit}^{(t)}_{5,j_2}\right)\cdot O(\epsilon\log d)\mathbb{1}_{\tau(x_1)=s}\right].$$

Turning to $\mathbf{Q}_{4,4}$, we have

$$\left[-\nabla_{\mathbf{Q}^{(\widetilde{T})}_{4,4}}\mathsf{Loss}^{2,2}_5\right]_{s,s}$$

$$= \mathbb{E}\Bigg[\mathbf{Attn}^{(\widetilde{T})}_{\mathsf{ans},1\to\mathsf{ans},1}\cdot\left(\left(1-\mathbf{logit}^{(\widetilde{T})}_{5,j_2}\right)\cdot\left(\left(V_{j_2,r_{g_2\cdot y_1}}(y_1)-\Lambda^{(\widetilde{T})}_{5,j_2,r_{g_2\cdot y_1}}\pm\widetilde{O}(\sigma_0)\right)\pm\widetilde{O}(\delta^q)\right)\right.$$

$$\left.\pm\frac{1}{\mathsf{poly}d}\left(1-\mathbf{logit}^{(\widetilde{T})}_{5,j_2}\right)\cdot\widetilde{O}(\sigma_0^q)\right)\mathbb{1}_{\{\tau(x_0)=s\}}\Bigg]$$

$$- \mathbb{E}\Bigg[\mathbf{Attn}^{(\widetilde{T})}_{\mathsf{ans},1\to\mathsf{ans},1}\cdot\left(\sum_{y\in\mathcal{Y}\setminus\{y_1\}}\mathbf{logit}^{(\widetilde{T})}_{5,\tau(g_2(y))}\cdot\mathbf{sReLU}'\big(\Lambda^{(\widetilde{T})}_{5,\tau(g_2(y)),r_{g_2\cdot y}}\big)\right.$$

$$\cdot \left( \left( V_{\tau(g_2(y)), r_{g_2 \cdot y}}(y_1) - \Lambda_{5,\tau(g_2(y)), r_{g_2 \cdot y}}^{(\widetilde{T})} \pm \widetilde{O}(\sigma_0) \right) \pm \widetilde{O}(\delta^q) \right) \right) \mathbb{1}_{\{\tau(x_0)=s\} \cap \widetilde{\mathcal{E}}_2} \right] \qquad (85)$$

$$- \mathbb{E} \left[ \mathbf{Attn}_{\mathsf{ans},1 \to \mathsf{ans},1}^{(t)} \cdot \sum_{j \neq j_2 \in \tau(\mathcal{Y})} \mathbf{logit}_{5,j}^{(t)} \cdot \right.$$

$$\left. \left( \sum_{r \in \widehat{\mathfrak{A}}_j} \mathbf{sReLU}'(\Lambda_{5,j,r}^{(t)}) \cdot \left( V_{j,r}(y_1) - \Lambda_{5,j,r}^{(t)} \pm \widetilde{O}(\sigma_0) \right) \pm \widetilde{O}(\delta^q) \right) \mathbb{1}_{\{\tau(x_1)=s\} \cap \widetilde{\mathcal{E}}_2^c} \right]. \qquad (86)$$

For (85), we have

$$(85) \geq \mathbb{E} \left[ \mathbf{Attn}_{\mathsf{ans},1 \to \mathsf{ans},1}^{(\widetilde{T})} \cdot \left( - \mathbf{logit}_{5,j_2'}^{(\widetilde{T})} \cdot \right. \right.$$

$$\left. \left. \min \left\{ \Omega(\log d), \Omega \left( ((\mathbf{Attn}_{\mathsf{ans},1 \to \mathsf{pred},2}^{(\widetilde{T})} - \mathbf{Attn}_{\mathsf{ans},1 \to \mathsf{ans},1}^{(\widetilde{T})}) \log d)^{q-1} \log d \right) \right\} \right) \mathbb{1}_{\{\tau(x_0)=s\} \cap \widetilde{\mathcal{E}}_2} \right]$$

$$\geq \mathbb{E} \left[ \left( 1 - \mathbf{logit}_{5,j_2}^{(\widetilde{T})} \right) \cdot \Omega \left( \frac{1}{d} \right) \cdot \right.$$

$$\left. \min \left\{ \Omega(\log d), \Omega \left( ((\mathbf{Attn}_{\mathsf{ans},1 \to \mathsf{pred},2}^{(\widetilde{T})} - \mathbf{Attn}_{\mathsf{ans},1 \to \mathsf{ans},1}^{(\widetilde{T})}) \log d)^{q-1} \log d \right) \right\} \mathbb{1}_{\{\tau(x_0)=s\} \cap \widetilde{\mathcal{E}}_2} \right].$$

Moreover, for (86), we have

$$(86) \geq - \mathbb{E} \left[ \mathbf{Attn}_{\mathsf{ans},1 \to \mathsf{ans},1}^{(\widetilde{T})} \cdot \mathbf{logit}_{5,\tau(g_1(y_1))}^{(\widetilde{T})} \cdot \mathbf{sReLU}' \left( \Lambda_{5,\tau(g_1(y_1)), r_{g_1 \cdot y_1}}^{(\widetilde{T})} \right) \right.$$

$$\left. \left( \left( V_{\tau(g_1(y_1)), r_{g_1 \cdot y_1}}(y_1) - \Lambda_{5,\tau(g_1(y_1)), r_{g_1 \cdot y_1}}^{(\widetilde{T})} \pm \widetilde{O}(\sigma_0) \right) \pm \widetilde{O}(\delta^q) \right) \mathbb{1}_{\{\tau(x_1)=s\} \cap \{g_1 \neq g_2 \wedge y_0 = y_1\}} \right]$$

$$\geq - \mathbb{E} \left[ \left( 1 - \mathbf{logit}_{5,j_2}^{(\widetilde{T})} \right) \cdot O \left( \frac{1}{d} \right) \cdot O \left( \frac{1}{\log d} \right) \cdot O \left( (\epsilon \log d)^{q-1} \right) \mathbb{1}_{\{\tau(x_0)=s\} \cap \widetilde{\mathcal{E}}_2} \right],$$

where the last inequality holds since $\Lambda_{5,\tau(g_1(y_1)), r_{g_1 \cdot y_1}}^{(\widetilde{T})} \leq \left( \mathbf{Attn}_{\mathsf{ans},1 \to \mathsf{pred},2}^{(\widetilde{T})} - \mathbf{Attn}_{\mathsf{ans},1 \to \mathsf{ans},1}^{(\widetilde{T})} \right) B \leq O(\epsilon \log d).$

Since $\mathbb{E} \left[ \Delta^{2,2} \mid \tau(x_1) = s \right] \geq \Omega \left( \frac{(d^{1.01} \epsilon)^{\frac{1}{q-1}}}{\log d} \right)$ we have

$$(85) \geq d^{0.01} \cdot \left| \left[ - \nabla_{\mathbf{Q}_{4,3}^{(\widetilde{T})}} \mathsf{Loss}_5^{2,2} \right]_{s,s} \right|,$$

and thus $(85) \gg (86)$. Thus, for $\epsilon \ll \frac{1}{d}$, if the attention gap does not decrease to the level of $O \left( \frac{(d^{1.01} \epsilon)^{\frac{1}{q-1}}}{\log d} \right)$, $[\mathbf{Q}_{4,4}]_{s,s}$ will start to dominantly increase while $[\mathbf{Q}_{4,3}]_{s,s}$ will not change too much. On the other hand, if the gap of attention holds, then $[\mathbf{Q}_{4,3}]_{s,s} \geq [\mathbf{Q}_{4,4}]_{s,s}$, we have

$$\epsilon_{\mathsf{attn}}^{2,2} = 1 - \mathbf{Attn}_{\mathsf{ans},1 \to \mathsf{pred},2}^{(t)} - \mathbf{Attn}_{\mathsf{ans},1 \to \mathsf{ans},1}^{(t)} = \frac{O(1)}{O(1) + e^{[\mathbf{Q}_{4,4}^{(t)}]_{s,s}} + e^{[\mathbf{Q}_{4,3}^{(t)}]_{s,s}}}$$

$$\geq \frac{O(1)}{O(1) + 2 e^{[\mathbf{Q}_{4,3}^{(\widetilde{T})}]_{s,s}}} \geq \frac{1}{2} \cdot 3\epsilon > \epsilon.$$

This implies that, we can find some time between $\widetilde{T}$ and $T_{2,3,s}$, s.t., the gap will decrease to $O \left( \frac{(d^{1.01} \epsilon)^{\frac{1}{q-1}}}{\log d} \right)$. We denote this time as $T^\star$ and stop the training.

### G.5.7 At the End of the Training

Putting everything together, we have that at the end of the training, we have

**Lemma G.45.** *Given $s \in \tau(\mathcal{X})$, at $T^\star = \widetilde{O}\left(\frac{d^{(1-2\epsilon)C_B}}{\eta\epsilon}\right)$, if $\widetilde{\Omega}(\sigma_0) \ll \epsilon \leq O(\frac{1}{d^{1.01}})$, we have*

(a) *Attention convergence: $\epsilon_{\mathsf{attn}}^{2,\ell} \leq O(\epsilon)$ for $\ell \in [2]$, and $\mathbb{E}\left[\Delta^{2,2} \mid \tau(x_1) = s\right] \leq O(\frac{(d^{1.01}\epsilon)^{\frac{1}{q-1}}}{\log d})$;*

(b) *$\left[\mathbf{Q}_{4,p}^{(T_{2,3,s})}\right]_{s,s'} \geq \Omega(\log d)$ for $p \in \{3,4\}$ if $s = s' \in \tau(\mathcal{X})$, otherwise, $\left[\mathbf{Q}_{4,p}^{(T_{2,3,s})}\right]_{s,s'} \leq \widetilde{O}(\frac{1}{d})$;*

(c) *Loss convergence: $\sum_{\ell=1}^{2} \mathsf{Loss}_5^{2,\ell} \leq \frac{1}{\mathsf{poly}d}$.*

### G.6 Proof of Main Theorem

**Theorem G.1** (Restatement of Theorem 4.1). *Under Assumptions 3.1, 3.2, 3.3, C.1, C.2 and 4.1, for some constant $0 < c^* < 1$, $n_y < m \ll \log^2 d$, the transformer model $F^{(T_1+T_2)}$ obtained by Algorithm 1 with learning rate $\eta = \frac{1}{\mathsf{poly}(d)}$, and stage 1 and 2 iteration $T_1 = \widetilde{O}\left(\frac{d}{\eta(\sigma_0)^{q-2}}\right)$, $T_2 = \widetilde{O}\left(\frac{\mathsf{poly}(d)}{\eta\sigma_0}\right)$ satisfies*

$$\mathrm{Acc}_L\left(F^{(T_1+T_2)}\right) \geq 1 - \frac{1}{\mathsf{poly}(d)}, \text{ for every } L \leq O(d^{c^*}), \tag{87}$$

*i.e., $F^{(T_1+T_2)}$, which is trained for task $\mathcal{T}^1$ and $\mathcal{T}^2$, generalizes to solve the tasks $\mathcal{T}^\ell, \ell \leq L$.*

*Proof.* By Lemma G.45, at the end of Stage 2 training, we have $\left[\mathbf{Q}_{4,p}^{(T_{2,3,s})}\right]_{s,s} \geq \Omega(\log d)$ for all $p \in \{3,4\}$ and $s \in \tau(\mathcal{X})$.

Therefore, for task $\mathcal{T}^L$ with $L \leq O(d^{c^*})$, where $0 < c^* < 1$ is a constant depending on the value of $\epsilon$ in Lemma G.45, we obtain

$$\epsilon_{\mathsf{attn}}^{L,\ell} \leq \frac{O(1) \cdot L}{O(1) \cdot L + e^{\left[\mathbf{Q}_{4,3}^{(T_{2,3,s})}\right]_{s,s}} + e^{\left[\mathbf{Q}_{4,4}^{(T_{2,3,s})}\right]_{s,s}}} = o(1).$$

Moreover, we have

$$\Delta^{L,\ell} \leq \Delta^{2,1} = \mathbf{Attn}_{\mathsf{ans},1\to\mathsf{pred},2}^{(T_{2,3,s})} - \mathbf{Attn}_{\mathsf{ans},1\to\mathsf{ans},1}^{(T_{2,3,s})} \leq o(1).$$

These together guarantee that

$$1 - \mathbf{logit}_{5,\tau(g_{\ell+1}(y_\ell))}(F^{(T^\star)}, \mathbf{Z}^{(L,\ell)}) \leq \frac{O(1) \cdot d + e^{o(1)}}{O(1) \cdot d + e^{o(1)} + e^{\Omega(\log d)}} \leq \frac{1}{\mathsf{poly}(d)},$$

which implies

$$\mathrm{Acc}_L\left(F^{(T^\star)}\right) \geq 1 - \frac{1}{\mathsf{poly}(d)}.$$

$\square$

## H Learning the Attention Layer: Symmetry Case for Short-Length

In this part, we consider the scenario where the group operations form a symmetry group. Specifically, following Assumption 5.1, we assume that $\mathcal{Y} = \{0, 1, \dots, n_y - 1\}$, and let $\mathcal{G}$ be the symmetry group for $\mathcal{Y}$, with order $|\mathcal{G}| = n_y! = \Theta(\mathsf{polylog}d) \gg \frac{1}{\varrho}$, where $n_y = \Theta\left(\frac{\log\log d}{\log\log\log d}\right)$. Similar to the simply transitive case, we restrict our analysis to updating only $\mathbf{Q}$, specifically the blocks $\mathbf{Q}_{4,3}$ and $\mathbf{Q}_{4,4}$.

Throughout this section, we focus on the simple task $\mathcal{T}^2$ and analyze gradient descent updates with respect to the per-token loss $\mathsf{Loss}_5^{2,2}$. Given $s \in \tau(\mathcal{X})$, let $\mathsf{Loss}_{5,s}^{2,2} = -\mathbb{E}\left[\log p_F(\mathbf{Z}_{\mathsf{ans},2,5} \mid \right.$

$\mathbf{Z}^{2,1}) \big| \tau(x_1) = s \big]$. Due to the symmetry of $[\mathbf{Q}_{4,3}]_{s,s}$ and $[\mathbf{Q}_{4,4}]_{s,s}$ across $s \in \tau(\mathcal{X})$, we may, without loss of generality, focus on a particular $s \in \tau(\mathcal{X})$ and analyze the corresponding loss $\mathsf{Loss}_{5,s}^{2,2}$ in what follows.

## H.1 Gradient Computations

We start with the gradient computations for the attention layer.

**Notations for gradient expressions.** We first introduce some notations for the gradients of the attention layer. For $1 \le \ell \le L$, given $\mathbf{Z}^{L,\ell-1}$ and $\mathbf{k} \in \mathcal{I}^{L,\ell-1}$, define

$$\Xi_{\ell,i,\mathbf{k}}^L(\mathbf{Z}^{L,\ell-1}) \triangleq \sum_{j\in[d]} \mathcal{E}_{i,j}(\mathbf{Z}^{L,\ell-1}) \sum_{r\in[m]} \mathbf{sReLU}'\big(\Lambda_{i,j,r}(\mathbf{Z}^{L,\ell-1})\big)\langle \mathbf{W}_{i,j,r}, \mathbf{Z}_{\mathbf{k}}\rangle, \quad i \in [5]. \quad (88)$$

For simplicity of notation, we will henceforth omit the dependence on $\mathbf{Z}^{L,\ell-1}$ in the notation of $\Xi_{\ell,i,\mathbf{k}}^L$ when it is clear from the context.

**Fact H.1** (Gradients of $\mathbf{Q}$). For any $p, q \in [5]$, we have

$$-\nabla_{\mathbf{Q}_{p,q}}\mathsf{Loss}^L = \sum_{\ell=1}^L \sum_{i\in[5]} -\nabla_{\mathbf{Q}_{p,q}}\mathsf{Loss}_i^{L,\ell}, \quad \text{where}$$

$$-\nabla_{\mathbf{Q}_{p,q}}\mathsf{Loss}_i^{L,\ell} =$$

$$\mathbb{E}\left[\sum_{\mathbf{k}\in\mathcal{I}^{L,\ell-1}} \mathbf{Attn}_{\mathsf{ans},\ell-1\to\mathbf{k}} \cdot \left(\Xi_{\ell,i,\mathbf{k}}^L - \sum_{\mathbf{k}'\in\mathcal{I}^{L,\ell-1}} \mathbf{Attn}_{\mathsf{ans},\ell-1\to\mathbf{k}'}\Xi_{\ell,i,\mathbf{k}'}^L\right)\mathbf{Z}_{\mathsf{ans},\ell-1,p}\mathbf{Z}_{\mathbf{k},q}^\top\right].$$

**Lemma H.1** (Gradients of $\mathbf{Q}_{4,3}$). *Given $s \in \tau(\mathcal{X})$, for the diagonal entry $[\mathbf{Q}_{4,3}]_{s,s}$ of the block $\mathbf{Q}_{4,3}$, we have*

$$\left[-\nabla_{\mathbf{Q}_{4,3}}\mathsf{Loss}_5^{2,2}\right]_{s,s} = \mathbb{E}\Bigg[\mathbf{Attn}_{\mathsf{ans},1\to\mathsf{pred},2} \cdot \Bigg(\sum_{j\in[d]} \mathcal{E}_{5,j}(\mathbf{Z}^{2,1}) \sum_{r\in[m]} \mathbf{sReLU}'\big(\Lambda_{5,j,r}\big)\cdot$$

$$\left(\langle\mathbf{W}_{5,j,r}, \mathbf{Z}_{\mathsf{pred},2}\rangle - \Lambda_{5,j,r} + b_{5,j,r}\right)\Bigg)\mathbb{1}_{s=\tau(x_1)}\Bigg].$$

*Moreover, for the off-diagonal entries $[\mathbf{Q}_{4,3}]_{s,s'}$ with $s \ne s'$, we have*

$$\left[-\nabla_{\mathbf{Q}_{4,3}}\mathsf{Loss}_5^{2,2}\right]_{s,s'} = \mathbb{E}\Bigg[\mathbf{Attn}_{\mathsf{ans},1\to\mathsf{pred},1} \cdot \Bigg(\sum_{j\in[d]} \mathcal{E}_{5,j}(\mathbf{Z}^{2,1}) \sum_{r\in[m]} \mathbf{sReLU}'\big(\Lambda_{5,j,r}\big)\cdot$$

$$\left(\langle\mathbf{W}_{5,j,r}, \mathbf{Z}_{\mathsf{pred},1}\rangle - \Lambda_{5,j,r} + b_{5,j,r}\right)\Bigg)\mathbb{1}_{s=\tau(x_1),s'=\tau(x_0)}\Bigg].$$

**Lemma H.2** (Gradients of $\mathbf{Q}_{4,4}$). *Given $s \in \tau(\mathcal{X})$, for the diagonal entry $[\mathbf{Q}_{4,4}]_{s,s}$ of the block $\mathbf{Q}_{4,4}$, we have*

$$\left[-\nabla_{\mathbf{Q}_{4,4}}\mathsf{Loss}_5^{2,2}\right]_{s,s} = \mathbb{E}\Bigg[\mathbf{Attn}_{\mathsf{ans},1\to\mathsf{ans},1} \cdot \Bigg(\sum_{j\in[d]} \mathcal{E}_{5,j}(\mathbf{Z}^{2,1}) \sum_{r\in[m]} \mathbf{sReLU}'\big(\Lambda_{5,j,r}\big)\cdot$$

$$\left(\langle\mathbf{W}_{5,j,r}, \mathbf{Z}_{\mathsf{ans},1}\rangle - \Lambda_{5,j,r} + b_{5,j,r}\right)\Bigg)\mathbb{1}_{s=\tau(x_1)}\Bigg].$$

*Moreover, for the off-diagonal entries $[\mathbf{Q}_{4,4}]_{s,s'}$ with $s \ne s'$, we have*

$$\left[-\nabla_{\mathbf{Q}_{4,4}}\mathsf{Loss}_5^{2,2}\right]_{s,s'} = \mathbb{E}\Bigg[\mathbf{Attn}_{\mathsf{ans},1\to\mathsf{ans},0} \cdot \Bigg(\sum_{j\in[d]} \mathcal{E}_{5,j}(\mathbf{Z}^{2,1}) \sum_{r\in[m]} \mathbf{sReLU}'\big(\Lambda_{5,j,r}\big)\cdot$$

$$\left(\left(\langle \mathbf{W}_{5,j,r}, \mathbf{Z}_{\mathsf{ans},0}\rangle - \Lambda_{5,j,r} + b_{5,j,r}\right)\right) \mathbb{1}_{s=\tau(x_1),s'=\tau(x_0)}\bigg].$$

## H.2 Some Useful Bounds for Gradients

In this subsection, we establish several useful bounds on the gradients of the attention layer, leveraging the feature structure of the MLP layer learned during stage 1.1. These bounds will be instrumental for the subsequent analysis.

Recall that for $j \in \tau(\mathcal{Y})$ and $y \in \mathcal{Y}$, the fiber $\mathrm{Fiber}_{j,y}$, and the set of feature combinations for predicting $y = \tau^{-1}(j)$, are defined as

$$\mathrm{Fiber}_{j,y} = \{g \in \mathcal{G} \mid \tau\big(g(y)\big) = j\}, \quad \mathfrak{F}_j = \left\{(\mathrm{Fiber}_{j,y}, y) \in 2^{\mathcal{G}} \times \mathcal{Y}\right\}.$$

As established in Theorem F.1, at the end of stage 1.1, we have

- **Sparse activations**: For $j \in \tau(\mathcal{Y})$, let $\phi = (\mathrm{Fiber}_{j,y}, y) \in \mathfrak{F}_j$, then there exists exactly one *activated* neuron $r \in [m]$ such that when $g_1 = g \in \mathrm{Fiber}_{j,y}, y_0 = y$ happens:

$$\Lambda_{5,j,r}^{(T_{1.1})} \geq B - O(\delta), \qquad \left|C_\alpha V_{j,r}^{(t)}(y) - V_{j,r}^{(t)}(g)\right| \leq O(\delta)$$

$$\Lambda_{5,j,r'}^{(T_{1.1})} \leq O(d^{-\Omega(1)}) \quad \forall r' \neq r.$$

  for some $C_\alpha = \Theta(n_y)$;

- **Cancellation of incorrect features**: For $j \in \tau(\mathcal{Y})$, let $\phi = (\mathrm{Fiber}_{j,y}, y) \in \mathfrak{F}_j$, and let the $r \in [m]$ be the activated neuron, then for any $g' \notin \mathrm{Fiber}_{j,y}$, and any $y' \neq y \in \mathcal{Y}$, we have

$$\left|V_{j,r}^{(T_{1.1})}(g) + V_{j,r}^{(T_{1.1})}(y')\right| \leq O(\delta) \quad \text{and} \quad \left|V_{j,r}^{(T_{1.1})}(g') + V_{j,r}^{(T_{1.1})}(y)\right| \leq O(\delta).$$

In the analysis of FFN layer, we have a pair $\delta = (\delta_1, \delta_2)$; in the expressions above, some terms should use $\delta_1$ and others $\delta_2$.

**Notations for activated neurons.** We denote by $r_{j,y}$ the unique activated neuron corresponding to $\phi = (\mathrm{Fiber}_{j,y}, y) \in \mathfrak{F}_j$. For any $g \in \mathrm{Fiber}_{j,y}$, we also write $r_{g \cdot y}$ for the same neuron $r_{j,y}$. Note that $r_{g_1 \cdot y} = r_{g_2 \cdot y}$ for distinct $g_1, g_2 \in \mathrm{Fiber}_{j,y}$. Moreover, define

$$\mathfrak{A} \triangleq \cup_{j \in \tau(\mathcal{Y})} \mathfrak{A}_j, \quad \text{where } \mathfrak{A}_j \triangleq \{r_{j,y} \mid y \in \mathcal{Y}\}.$$

In other words, $\mathfrak{A}$ is the set of all activated neurons across all feature sets $\mathfrak{F}_j$ for $j \in \tau(\mathcal{Y})$. Given $\mathbf{Z}^{L,\ell-1}$, letting $\widehat{\mathcal{G}}(\mathbf{Z}^{L,\ell-1}) = \cup \{g_{\ell'}\}_{\ell'=1}^{L}$ be the collection of all the chosen group elements in the predicate clauses. Similarly $\widehat{\mathcal{Y}} = \cup \{y_{\ell'}\}_{\ell'=0}^{\ell-1}$. Then define $\widehat{\mathfrak{A}}_j(\mathbf{Z}^{L,\ell-1}) = \left\{r_{g \cdot y} \mid g \in \mathrm{Fiber}_{j,y} \wedge \right.$

$\left. \left(g \in \widehat{\mathcal{G}}(\mathbf{Z}^{L,\ell-1}) \vee y \in \widehat{\mathcal{Y}}\right)\right\}$. For simplicity, we omit the dependence on $\mathbf{Z}^{L,\ell-1}$ in the notation of $\widehat{\mathfrak{A}}_j$ when it is clear from the context. Equipped with these notations, we can summarize the above properties in the following lemmas.

**Lemma H.3** (Properties of target feature magnitude). *Given $j \in \tau(\mathcal{Y})$ and $y \in \mathcal{Y}$ then for $g \in \mathrm{Fiber}_{j,y}$, the following properties hold.*

$$V_{j,r_{g \cdot y}}(g) + V_{j,r_{g \cdot y}}(y) \geq 2B - O(\delta), \quad \left|C_\alpha V_{j,r}^{(t)}(y) - V_{j,r}^{(t)}(g)\right| \leq O(\delta), \tag{89}$$

$$\left|V_{j,r_{g \cdot y}}(g) + V_{j,r_{g \cdot y}}(y')\right| \leq O(\delta), V_{j,r_{g \cdot y}}(y') < 0 \quad \text{for all } y' \neq y, \tag{90}$$

$$\left|V_{j,r_{g \cdot y}}(g') + V_{j,r_{g \cdot y}}(y)\right| \leq O(\delta), V_{j,r_{g \cdot y}}(g') < 0 \quad \text{for all } g' \notin \mathrm{Fiber}_{j,y}. \tag{91}$$

$$|V_{j,r}(g)|, |V_{j,r}(y)| \leq O(\delta) \quad \text{for all } r \notin \mathfrak{A}_j. \tag{92}$$

**Lemma H.4** (Properties of irrelevant magnitude). *If $(p,v) \notin \{2\} \times \mathcal{G} \cup \{5\} \times \mathcal{Y}$, or $j \notin \tau(\mathcal{Y})$, then for any $r \in [m]$, we have*

$$\left|\langle \mathbf{W}_{5,j,r,p}, e_v\rangle\right| \leq \widetilde{O}(\sigma_0). \tag{93}$$

The above lemmas give us some direct computations of the inner products between the weight matrices and input embedding vectors.

**Lemma H.5.** *Let $j \in \tau(\mathcal{Y})$ and $\ell \in [2]$. Then for any $r \in [m]$, the following holds:*

$$\langle \mathbf{W}_{5,j,r}, \mathbf{Z}_{\mathsf{pred},\ell} \rangle = V_{j,r}(g_\ell) \pm \widetilde{O}(\sigma_0), \tag{94}$$

$$\langle \mathbf{W}_{5,j,r}, \mathbf{Z}_{\mathsf{ans},\ell-1} \rangle = V_{j,r}(y_{\ell-1}) \pm \widetilde{O}(\sigma_0). \tag{95}$$

*Moreover, for $j \notin \tau(\mathcal{Y})$ and any $\mathbf{k} \in \mathcal{I}^{2,1}$ and $r \in [m]$, we have*

$$\left| \langle \mathbf{W}_{5,j,r}, \mathbf{Z}_{\mathbf{k}} \rangle \right| = \widetilde{O}(\sigma_0). \tag{96}$$

Furthermore, we can establish some characterizations of the $\Lambda_{5,j,r}(\mathbf{Z}^{2,\ell-1})$ quantities, which are crucial for the following analysis.

**Lemma H.6** (Characterizations of Lambda). *Given $\mathbf{Z}^{2,1}$ with $\{\mathbf{Attn}_{\mathsf{ans},1\to\mathbf{k}}\}_{\mathbf{k}\in\mathcal{I}^{2,1}}$, then*

*(a) for $j \in \tau(\mathcal{Y})$, for activated neuron $r \in \mathfrak{A}_j$, we have*

$$\Lambda_{5,j,r} = \sum_{\ell'=1}^{2} \mathbf{Attn}_{\mathsf{ans},1\to\mathsf{pred},\ell'} V_{j,r}(g_{\ell'}) + \sum_{\ell'=1}^{2} \mathbf{Attn}_{\mathsf{ans},1\to\mathsf{ans},\ell'-1} V_{j,r}(y_{\ell'-1}) \pm \widetilde{O}(\sigma_0).$$

*(b) for $j \in \tau(\mathcal{Y})$, for any non-activated neuron $r \notin \mathfrak{A}_j$ we have*

$$\left| \Lambda_{5,j,r} \right| \leq O(\delta).$$

*(c) for $j \notin \tau(\mathcal{Y})$, for any $r \in [m]$, we have*

$$\left| \Lambda_{5,j,r} \right| \leq \widetilde{O}(\sigma_0).$$

A direct consequence of the above lemma is the following finer characterization of the activated neurons.

**Lemma H.7.** *Given $j \in \tau(\mathcal{Y})$, for $r \in \mathfrak{A}_j \setminus \widehat{\mathfrak{A}}_j$, we have $\mathbf{sReLU}'(\Lambda_{5,j,r}) = 0$.*

Now we are ready to further derive the gradients of the attention layer starting from Lemmas H.1 and H.2 and the properties established above.

**Lemma H.8** (Refined expression for the gradient of $\mathbf{Q}_{4,3}$). *Given $s \in \tau(\mathcal{X})$, for the diagonal entry $[\mathbf{Q}_{4,3}]_{s,s}$ of the block $\mathbf{Q}_{4,3}$, letting $j_2 = \tau(g_2(y_1))$, we have*

$$\left[ -\nabla_{\mathbf{Q}_{4,3}} \mathsf{Loss}_5^{2,2} \right]_{s,s} = \mathbb{E}\Bigg[ \mathbf{Attn}_{\mathsf{ans},1\to\mathsf{pred},2} \cdot$$

$$\left( (1 - \mathbf{logit}_{5,j_2}) \cdot \Big( \sum_{r \in \widehat{\mathfrak{A}}_{j_2}} \mathbf{sReLU}'(\Lambda_{5,j_2,r}) \cdot \Big( V_{j_2,r}(g_2) - \Lambda_{5,j_2,r} \pm \widetilde{O}(\sigma_0) \Big) \pm \widetilde{O}(\delta^q) \Big) \right.$$

$$\left. - \sum_{j \neq j_2 \in \tau(\mathcal{Y})} \mathbf{logit}_{5,j} \cdot \Big( \sum_{r \in \widehat{\mathfrak{A}}_j} \mathbf{sReLU}'(\Lambda_{5,j,r}) \cdot \Big( V_{j,r}(g_2) - \Lambda_{5,j,r} \pm \widetilde{O}(\sigma_0) \Big) \pm \widetilde{O}(\delta^q) \Big) \right.$$

$$\left. \pm \sum_{j \notin \tau(\mathcal{Y})} \mathbf{logit}_{5,j} \widetilde{O}(\sigma_0^q) \Big) \mathbb{1}_{\tau(x_1)=s} \right].$$

*Moreover, for the off-diagonal entries $[\mathbf{Q}_{4,3}]_{s,s'}$ with $s \neq s'$, we have*

$$\left[ -\nabla_{\mathbf{Q}_{4,3}} \mathsf{Loss}_5^{2,2} \right]_{s,s'} = \mathbb{E}\Bigg[ \mathbf{Attn}_{\mathsf{ans},1\to\mathsf{pred},1} \cdot$$

$$\left( (1 - \mathbf{logit}_{5,j_2}) \cdot \Big( \sum_{r \in \widehat{\mathfrak{A}}_{j_2}} \mathbf{sReLU}'(\Lambda_{5,j_2,r}) \cdot \Big( V_{j_2,r}(g_1) - \Lambda_{5,j_2,r} \pm \widetilde{O}(\sigma_0) \Big) \pm \widetilde{O}(\delta^q) \Big) \right.$$

$$-\sum_{j\neq j_2\in\tau(\mathcal{Y})}\mathbf{logit}_{5,j}\cdot\Big(\sum_{r\in\widehat{\mathfrak{A}}_j}\mathbf{sReLU}'(\Lambda_{5,j,r})\cdot\Big(V_{j,r}(g_1)-\Lambda_{5,j,r}\pm\widetilde{O}(\sigma_0)\Big)\pm\widetilde{O}(\delta^q)\Big)$$

$$\pm\sum_{j\notin\tau(\mathcal{Y})}\mathbf{logit}_{5,j}\widetilde{O}(\sigma_0^q)\Big)\mathbb{1}_{\tau(x_1)=s,\tau(x_0)=s'}\Bigg].$$

**Lemma H.9** (Refined expression for the gradient of $\mathbf{Q}_{4,4}$). *Given $s\in\tau(\mathcal{X})$, for the diagonal entry $[\mathbf{Q}_{4,4}]_{s,s}$ of the block $\mathbf{Q}_{4,4}$, letting $j_2=\tau(g_2(y_1))$, we have*

$$\Big[-\nabla_{\mathbf{Q}_{4,4}}\mathsf{Loss}_5^{2,2}\Big]_{s,s}=\mathbb{E}\Bigg[\mathbf{Attn}_{\mathsf{ans},1\to\mathsf{ans},1}\cdot$$

$$\Big((1-\mathbf{logit}_{5,j_2})\cdot\Big(\sum_{r\in\widehat{\mathfrak{A}}_{j_2}}\mathbf{sReLU}'(\Lambda_{5,j_2,r})\cdot\Big(V_{j_2,r}(y_1)-\Lambda_{5,j_2,r}\pm\widetilde{O}(\sigma_0)\Big)\pm\widetilde{O}(\delta^q)\Big)$$

$$-\sum_{j\neq j_2\in\tau(\mathcal{Y})}\mathbf{logit}_{5,j}\cdot\Big(\sum_{r\in\widehat{\mathfrak{A}}_j}\mathbf{sReLU}'(\Lambda_{5,j,r})\cdot\Big(V_{j,r}(y_1)-\Lambda_{5,j,r}\pm\widetilde{O}(\sigma_0)\Big)\pm\widetilde{O}(\delta^q)\Big)$$

$$\pm\sum_{j\notin\tau(\mathcal{Y})}\mathbf{logit}_{5,j}\widetilde{O}(\sigma_0^q)\Big)\mathbb{1}_{\tau(x_1)=s}\Bigg].$$

*Moreover, for the off-diagonal entries $[\mathbf{Q}_{4,3}]_{s,s'}$ with $s\neq s'$, we have*

$$\Big[-\nabla_{\mathbf{Q}_{4,4}}\mathsf{Loss}_5^{2,2}\Big]_{s,s'}=\mathbb{E}\Bigg[\mathbf{Attn}_{\mathsf{ans},1\to\mathsf{ans},0}\cdot$$

$$\Big((1-\mathbf{logit}_{5,j_2})\cdot\Big(\sum_{r\in\widehat{\mathfrak{A}}_{j_2}}\mathbf{sReLU}'(\Lambda_{5,j_2,r})\cdot\Big(V_{j_2,r}(y_0)-\Lambda_{5,j_2,r}\pm\widetilde{O}(\sigma_0)\Big)\pm\widetilde{O}(\delta^q)\Big)$$

$$-\sum_{j\neq j_2\in\tau(\mathcal{Y})}\mathbf{logit}_{5,j}\cdot\Big(\sum_{r\in\widehat{\mathfrak{A}}_j}\mathbf{sReLU}'(\Lambda_{5,j,r})\cdot\Big(V_{j,r}(y_0)-\Lambda_{5,j,r}\pm\widetilde{O}(\sigma_0)\Big)\pm\widetilde{O}(\delta^q)\Big)$$

$$\pm\sum_{j\notin\tau(\mathcal{Y})}\mathbf{logit}_{5,j}\widetilde{O}(\sigma_0^q)\Big)\mathbb{1}_{\tau(x_1)=s,\tau(x_0)=s'}\Bigg].$$

Following the above calculations, we can further obtain the gradient summation of $\mathbf{Q}_{4,3}$ and $\mathbf{Q}_{4,4}$ as follows:

**Lemma H.10** (Gradient sum of $\mathbf{Q}_{4,3}$ and $\mathbf{Q}_{4,4}$). *Given $s\in\tau(\mathcal{X})$, letting $j_2=\tau(g_2(y_1))$, we have*

$$\Big[-\nabla_{\mathbf{Q}_{4,3}}\mathsf{Loss}_5^{2,2}\Big]_{s,s}+\Big[-\nabla_{\mathbf{Q}_{4,3}}\mathsf{Loss}_5^{2,2}\Big]_{s,s}$$

$$=\mathbb{E}\Bigg[\Big((1-\mathbf{logit}_{5,j_2})\cdot\Big(\sum_{r\in\widehat{\mathfrak{A}}_{j_2}}\mathbf{sReLU}'(\Lambda_{5,j_2,r})\cdot$$

$$\Big(-\mathbf{Attn}_{\mathsf{ans},1\to\mathsf{ans},0}\cdot V_{j_2,r}(y_0)-\mathbf{Attn}_{\mathsf{ans},1\to\mathsf{pred},1}\cdot V_{j_2,r}(g_1)$$

$$+\big(1-\mathbf{Attn}_{\mathsf{ans},1\to\mathsf{ans},1}-\mathbf{Attn}_{\mathsf{ans},1\to\mathsf{pred},2}\big)\Lambda_{5,j_2,r}\pm\widetilde{O}(\sigma_0)\Big)\pm\widetilde{O}(\delta^q)\Big)$$

$$+\sum_{j\neq j_2\in\tau(\mathcal{Y})}\mathbf{logit}_{5,j}\cdot\Big(\sum_{r\in\widehat{\mathfrak{A}}_j}\mathbf{sReLU}'(\Lambda_{5,j,r})\cdot$$

$$\Big(\mathbf{Attn}_{\mathsf{ans},1\to\mathsf{ans},0}\cdot V_{j,r}(y_0)+\mathbf{Attn}_{\mathsf{ans},1\to\mathsf{pred},1}\cdot V_{j,r}(g_1)$$

$$-\big(1-\mathbf{Attn}_{\mathsf{ans},1\to\mathsf{ans},1}-\mathbf{Attn}_{\mathsf{ans},1\to\mathsf{pred},2}\big)\Lambda_{5,j,r}\pm\widetilde{O}(\sigma_0)\Big)\pm\widetilde{O}(\delta^q)\Big)$$

$$\pm \sum_{j \notin \tau(\mathcal{Y})} \mathbf{logit}_{5,j} \cdot \left(\mathbf{Attn}_{\mathsf{ans},1 \to \mathsf{ans},1} + \mathbf{Attn}_{\mathsf{ans},1 \to \mathsf{pred},2}\right) \widetilde{O}(\sigma_0^q)\right) \mathbb{1}_{\tau(x_1)=s}\Bigg].$$

**Notations for gradient decompositions.** We shall define some useful notations to further simplify the expressions of gradient.

**Lemma H.11.** *For any $s \in \tau(\mathcal{X})$, we define the following notations for the gradient decompositions:*

1. *for $[\mathbf{Q}_{4,3}]_{s,s}$ we have $\left[-\nabla_{\mathbf{Q}_{4,3}} \mathsf{Loss}_5^{2,2}\right]_{s,s} = \mathbb{E}\left[\mathcal{N}_{s,3,2,i} + \mathcal{N}_{s,3,2,ii} + \mathcal{N}_{s,3,2,iii}\right]$, where*

$$\mathcal{N}_{s,3,2,i} = \mathbf{Attn}_{\mathsf{ans},1 \to \mathsf{pred},2} \cdot (1 - \mathbf{logit}_{5,j_2}) \cdot \tag{97}$$
$$\left(\sum_{r \in \widehat{\mathfrak{A}}_{j_2}} \mathbf{sReLU}'(\Lambda_{5,j_2,r}) \cdot \left(V_{j_2,r}(g_2) - \Lambda_{5,j_2,r} \pm \widetilde{O}(\sigma_0)\right) \pm \widetilde{O}(\delta^q)\right) \mathbb{1}_{\tau(x_1)=s},$$

$$\mathcal{N}_{s,3,2,ii} = -\mathbf{Attn}_{\mathsf{ans},1 \to \mathsf{pred},2} \cdot \sum_{j \neq j_2 \in \tau(\mathcal{Y})} \mathbf{logit}_{5,j} \cdot \tag{98}$$
$$\left(\sum_{r \in \widehat{\mathfrak{A}}_j} \mathbf{sReLU}'(\Lambda_{5,j,r}) \cdot \left(V_{j,r}(g_2) - \Lambda_{5,j,r} \pm \widetilde{O}(\sigma_0)\right) \pm \widetilde{O}(\delta^q)\right) \mathbb{1}_{\tau(x_1)=s},$$

$$\mathcal{N}_{s,3,2,iii} = \pm \mathbf{Attn}_{\mathsf{ans},1 \to \mathsf{pred},2} \cdot \sum_{j \notin \tau(\mathcal{Y})} \mathbf{logit}_{5,j} \widetilde{O}(\sigma_0^q) \mathbb{1}_{\tau(x_1)=s}. \tag{99}$$

2. *for $[\mathbf{Q}_{4,4}]_{s,s}$, we have $\left[-\nabla_{\mathbf{Q}_{4,4}} \mathsf{Loss}_5^{2,2}\right]_{s,s} = \mathbb{E}\left[\mathcal{N}_{s,4,2,i} + \mathcal{N}_{s,4,2,ii} + \mathcal{N}_{s,4,2,iii}\right]$, where*

$$\mathcal{N}_{s,4,2,i} = \mathbf{Attn}_{\mathsf{ans},1 \to \mathsf{ans},1} \cdot (1 - \mathbf{logit}_{5,j_2}) \cdot \tag{100}$$
$$\left(\sum_{r \in \widehat{\mathfrak{A}}_{j_2}} \mathbf{sReLU}'(\Lambda_{5,j_2,r}) \cdot \left(V_{j_2,r}(y_1) - \Lambda_{5,j_2,r} \pm \widetilde{O}(\sigma_0)\right) \pm \widetilde{O}(\delta^q)\right) \mathbb{1}_{\tau(x_1)=s},$$

$$\mathcal{N}_{s,4,2,ii} = -\mathbf{Attn}_{\mathsf{ans},1 \to \mathsf{ans},1} \cdot \sum_{j \neq j_2 \in \tau(\mathcal{Y})} \mathbf{logit}_{5,j} \cdot \tag{101}$$
$$\left(\sum_{r \in \widehat{\mathfrak{A}}_j} \mathbf{sReLU}'(\Lambda_{5,j,r}) \cdot \left(V_{j,r}(y_1) - \Lambda_{5,j,r} \pm \widetilde{O}(\sigma_0)\right) \pm \widetilde{O}(\delta^q)\right) \mathbb{1}_{\tau(x_1)=s},$$

$$\mathcal{N}_{s,4,2,iii} = \pm \mathbf{Attn}_{\mathsf{ans},1 \to \mathsf{ans},1} \cdot \sum_{j \notin \tau(\mathcal{Y})} \mathbf{logit}_{5,j} \widetilde{O}(\sigma_0^q) \mathbb{1}_{\tau(x_1)=s}. \tag{102}$$

3. *for the summation of $[\mathbf{Q}_{4,3}]_{s,s}$ and $[\mathbf{Q}_{4,4}]_{s,s}$, we have $\left[-\nabla_{\mathbf{Q}_{4,3}} \mathsf{Loss}_5^{2,2}\right]_{s,s} + \left[-\nabla_{\mathbf{Q}_{4,4}} \mathsf{Loss}_5^{2,2}\right]_{s,s} = \mathbb{E}\left[\mathcal{N}_{s,2,i} + \mathcal{N}_{s,2,ii} + \mathcal{N}_{s,2,iii}\right]$, where*

$$\mathcal{N}_{s,2,i} = (1 - \mathbf{logit}_{5,j_2}) \cdot \left(\sum_{r \in \widehat{\mathfrak{A}}_{j_2}} \mathbf{sReLU}'(\Lambda_{5,j_2,r}) \cdot \right.$$
$$\left(-\mathbf{Attn}_{\mathsf{ans},1 \to \mathsf{ans},0} \cdot V_{j_2,r}(y_0) - \mathbf{Attn}_{\mathsf{ans},1 \to \mathsf{pred},1} \cdot V_{j_2,r}(g_1)\right.$$
$$\left.+ \left(1 - \mathbf{Attn}_{\mathsf{ans},1 \to \mathsf{ans},1} - \mathbf{Attn}_{\mathsf{ans},1 \to \mathsf{pred},2}\right)\Lambda_{5,j_2,r} \pm \widetilde{O}(\sigma_0)\right) \pm \widetilde{O}(\delta^q)\right) \mathbb{1}_{\tau(x_1)=s}$$
$$\mathcal{N}_{s,2,ii} = \sum_{j \neq j_2 \in \tau(\mathcal{Y})} \mathbf{logit}_{5,j} \cdot \left(\sum_{r \in \widehat{\mathfrak{A}}_j} \mathbf{sReLU}'(\Lambda_{5,j,r}) \cdot \right.$$
$$\left(\mathbf{Attn}_{\mathsf{ans},1 \to \mathsf{ans},0} \cdot V_{j,r}(y_0) + \mathbf{Attn}_{\mathsf{ans},1 \to \mathsf{pred},1} \cdot V_{j,r}(g_1)\right.$$
$$\left.- \left(1 - \mathbf{Attn}_{\mathsf{ans},1 \to \mathsf{ans},1} - \mathbf{Attn}_{\mathsf{ans},1 \to \mathsf{pred},2}\right)\Lambda_{5,j,r} \pm \widetilde{O}(\sigma_0)\right) \pm \widetilde{O}(\delta^q)\right) \mathbb{1}_{\tau(x_1)=s}$$

$$\mathcal{N}_{s,2,iii} = \pm \sum_{j \notin \tau(\mathcal{Y})} \mathbf{logit}_{5,j} \cdot \left(\mathbf{Attn}_{\mathsf{ans},1\to\mathsf{ans},1} + \mathbf{Attn}_{\mathsf{ans},1\to\mathsf{pred},2}\right) \widetilde{O}(\sigma_0^q) \mathbb{1}_{\tau(x_1)=s}.$$

**Probabilistic events.** We conclude this subsection by introducing several probabilistic events that will be used to simplify the characterization of activated neurons in the subsequent analysis. We first define some events that may contribute non-trivially to the gradient of $\mathbf{Q}_{4,3}$ and $\mathbf{Q}_{4,4}$:

$$\mathcal{E}_1 = \left\{ y_0 \neq y_1, g_1(y_{\ell-1}) \neq g_2(y_{\ell-1}), \text{ for all } \ell \in [2] \right\}, \tag{103}$$

$$\mathcal{E}_2 = \left\{ g_1, g_2 \in \mathrm{Fiber}_{\tau(g_2(y_0)),y_0}, g_1(y_1) \neq g_2(y_1) \right\}, \tag{104}$$

$$\mathcal{E}_3 = \left\{ y_0 = y_1, g_1(y_1) \neq g_2(y_1) \right\}, \tag{105}$$

$$\mathcal{E}_4 = \left\{ y_0 \neq y_1, g_1(y_{\ell-1}) = g_2(y_{\ell-1}), \text{ for all } \ell \in [2] \right\}. \tag{106}$$

We first observe that event $\mathcal{E}_1$ occurs with high probability $1 - O\left(\frac{1}{n_c}\right)$, and serves as the primary regime of interest. In contrast, events $\mathcal{E}_2$ and $\mathcal{E}_3$ each occur with probability $\Theta(1/n_y)$ and correspond to instances of initial prediction ambiguity, where certain incorrect classes may exhibit disproportionately large logits. Similarly, $\mathcal{E}_4$ corresponds to instances of initial prediction ambiguity, while occurs with probability $\Theta(1/n_y^2)$. Furthermore, we define the following events, which yield negligible gradient contributions to $\mathbf{Q}_{4,3}$ and $\mathbf{Q}_{4,4}$, as they do not lead to significant confusion among the incorrect predictions:

$$\mathcal{E}_5 = \left\{ y_0 \neq y_1, g_1(y_1) = g_2(y_1), g_1(y_0) \neq g_2(y_0) \right\}, \tag{107}$$

$$\mathcal{E}_6 = \left\{ y_0 = y_1, g_1(y_1) = g_2(y_1) \right\}. \tag{108}$$

Here, $\mathcal{E}_5$ occurs with probability $\Theta(1/n_y)$, and $\mathcal{E}_6$ occurs with probability $\Theta(1/n_c^2)$. Together, the events $\cup_{i \in [6]} \mathcal{E}_i$ form a partition of the entire sample space.

### H.3 Stage 1.2.1: Initial Growth of Q

We define the following notations:

$$\epsilon_{\mathsf{attn}}^{L,\ell}\left(\mathbf{Z}^{L,\ell-1}\right) = 1 - \mathbf{Attn}_{\mathsf{ans},\ell-1\to\mathsf{pred},\ell}\left(\mathbf{Z}^{L,\ell-1}\right) - \mathbf{Attn}_{\mathsf{ans},\ell-1\to\mathsf{ans},\ell-1}\left(\mathbf{Z}^{L,\ell-1}\right),$$

$$\Delta^{L,\ell}\left(\mathbf{Z}^{L,\ell-1}\right) = \left|\mathbf{Attn}_{\mathsf{ans},\ell-1\to\mathsf{pred},\ell}\left(\mathbf{Z}^{L,\ell-1}\right) - \mathbf{Attn}_{\mathsf{ans},\ell-1\to\mathsf{ans},\ell-1}\left(\mathbf{Z}^{L,\ell-1}\right)\right|.$$

We abbreviate $\epsilon_{\mathsf{attn}}^{L,\ell}\left(\mathbf{Z}^{L,\ell-1}\right)$ and $\Delta^{L,\ell}\left(\mathbf{Z}^{L,\ell-1}\right)$ as $\epsilon_{\mathsf{attn}}^{L,\ell}$ and $\Delta^{L,\ell}$ for simplicity. Since we only focus on the input $\mathbf{Z}^{2,1}$ in the following analysis, we will omit the notation related to the length, i.e., we use $\epsilon_{\mathsf{attn}}$ and $\Delta$ when the context is clear.

**Induction H.1.** *Given $s \in \tau(\mathcal{X})$, let $T_{1,2,1,s}$ denote the first time that $\mathbb{E}[\epsilon_{\mathsf{attn}} \mid \tau(x_1) = s] \leq 0.4$ For all iterations $t \leq T_{1,2,1,s}$, we have the following holds*

(a) $\left[\mathbf{Q}_{4,3}^{(t)}\right]_{s,s} + \left[\mathbf{Q}_{4,4}^{(t)}\right]_{s,s} \leq O(1)$ *monotonically increases;*

(b) *for $p \in \{3,4\}$, for $s' \in \tau(\mathcal{X}) \neq s$, $\left|\left[\mathbf{Q}_{4,p}^{(t)}\right]_{s,s'}\right| \leq O\left(\frac{\left[\mathbf{Q}_{4,p}^{(t)}\right]_{s,s'}}{d}\right)$ ; otherwise $\left[\mathbf{Q}_{4,p}^{(t)}\right]_{s,s'} = 0$;*

(c) *for any sample $\mathbf{Z}^{2,1}$, we have $\mathbf{Attn}_{\mathsf{ans},1\to\mathsf{pred},2}^{(t)} - \mathbf{Attn}_{\mathsf{ans},1\to\mathsf{pred},1}^{(t)} \geq -O\left(\frac{\log\log d}{\log d}\right)$;*

(d) *for any sample $\mathbf{Z}^{2,1}$, we have $\mathbf{Attn}_{\mathsf{ans},1\to\mathsf{pred},2}^{(t)} - \mathbf{Attn}_{\mathsf{ans},1\to\mathsf{ans},1}^{(t)} \leq c_1$ for some small constant $c_1 > 0$.*

#### H.3.1 Attention and Lambda Preliminaries

**Lemma H.12.** *If Induction H.1 holds for all iterations $< t$, then we have*

1. $\mathbf{Attn}_{\mathsf{ans},1\to\mathsf{pred},1}^{(t)} + \mathbf{Attn}_{\mathsf{ans},1\to\mathsf{ans},1}^{(t)} \in [0.4 \pm \widetilde{O}\left(\frac{1}{d}\right), 0.5]$;

2. $\left|\mathbf{Attn}_{\mathsf{ans},1\to\mathsf{pred},1}^{(t)} - \mathbf{Attn}_{\mathsf{ans},1\to\mathsf{ans},0}^{(t)}\right| \leq \widetilde{O}\left(\frac{1}{d}\right)$.

**Lemma H.13.** *If Induction H.1 holds for all iterations $< t$, then given $\mathbf{Z}^{2,1} \in \mathcal{E}_1$,*

*1. for the prediction $j_2$, we have*

$$\Lambda^{(t)}_{5,j_2,r_{g_2 \cdot y_1}} =$$
$$\left(\mathbf{Attn}^{(t)}_{\mathsf{ans},1\to\mathsf{pred},2} - \mathbf{Attn}^{(t)}_{\mathsf{ans},1\to\mathsf{ans},0}\right) \cdot 2B + \left(\mathbf{Attn}^{(t)}_{\mathsf{ans},1\to\mathsf{ans},1} - \mathbf{Attn}^{(t)}_{\mathsf{ans},1\to\mathsf{pred},1}\right) \cdot \frac{2B}{n_y} + O(\delta).$$

*2. for the prediction $j_2' = \tau\big(g_2(y_0)\big)$, we have*

$$\Lambda^{(t)}_{5,j_2',r_{g_2 \cdot y_0}} = \left(\mathbf{Attn}^{(t)}_{\mathsf{ans},1\to\mathsf{pred},2} - \mathbf{Attn}^{(t)}_{\mathsf{ans},1\to\mathsf{ans},1}\right) \cdot 2B + \widetilde{O}\left(\frac{B}{d \cdot n_y}\right) + O(\delta).$$

*3. for the prediction $\tau(g_1(y_0))$, we have*

$$\Lambda^{(t)}_{5,\tau(g_1(y_0)),r_{g_1 \cdot y_0}} =$$
$$\left(\mathbf{Attn}^{(t)}_{\mathsf{ans},1\to\mathsf{pred},1} - \mathbf{Attn}^{(t)}_{\mathsf{ans},1\to\mathsf{ans},1}\right) \cdot 2B + \left(\mathbf{Attn}^{(t)}_{\mathsf{ans},1\to\mathsf{ans},0} - \mathbf{Attn}^{(t)}_{\mathsf{ans},1\to\mathsf{pred},2}\right) \cdot \frac{2B}{n_y} + O(\delta).$$

*furthermore, we have $\Lambda^{(t)}_{5,\tau(g_1(y_0)),r_{g_1 \cdot y_0}} \le \Lambda^{(t)}_{5,\tau(g_2(y_1)),r_{g_2 \cdot y_1}}$.*

*4. for the prediction $\tau(g_1(y_1))$, we have*

$$\Lambda^{(t)}_{5,\tau(g_1(y_1)),r_{g_1 \cdot y_1}} = \left(\mathbf{Attn}^{(t)}_{\mathsf{ans},1\to\mathsf{ans},1} - \mathbf{Attn}^{(t)}_{\mathsf{ans},1\to\mathsf{pred},2}\right) \cdot \frac{2B}{n_y} + \widetilde{O}\left(\frac{B}{d}\right) + O(\delta).$$

*5. for other $j \in \tau(\mathcal{Y})$, if there are $j$ and $y$, s.t., $g_1, g_2 \in \mathrm{Fiber}_{j,y}$ (notice that $y \ne y_0, y_1$ for $\mathcal{E}_1$), then for such a $j$, $r \in \widehat{\mathfrak{A}}_j \setminus \{r_{g_2 \cdot y}\}$ cannot be activated, moreover, we have*

$$\Lambda^{(t)}_{5,j,r_{g_2 \cdot y}} = \left(\mathbf{Attn}^{(t)}_{\mathsf{ans},1\to\mathsf{pred},2} - \mathbf{Attn}^{(t)}_{\mathsf{ans},1\to\mathsf{ans},1}\right) \cdot 2B + \widetilde{O}\left(\frac{B}{d \cdot n_y}\right) + O(\delta);$$

*else, none of $r \in \widehat{\mathfrak{A}}_j$ can be activated.*

*Notice that for $\mathcal{E}_1$, the neurons mentioned in 1-4 should be different neurons, while the predictions in 1 and 3, or 2 and 4 can be the same in some cases, e.g., $g_1(y_1) = g_2(y_0)$. In all cases, except for the neurons mentioned above, i.e., $\cup_{\ell,\ell' \in [2]}\{r_{g_\ell \cdot y_{\ell'-1}}\}$ (which may not be activated), all other neurons $r \in \cup_{\ell,\ell' \in [2]}\left(\widehat{\mathfrak{A}}_{\tau(g_\ell \cdot y_{\ell'-1})} \setminus \{r_{g_\ell \cdot y_{\ell'-1}}\}\right)$ cannot be activated.*

**Lemma H.14.** *If Induction H.1 holds for all iterations $< t$, then given $\mathbf{Z}^{2,1} \in \mathcal{E}_2$, we have*

*1. for the prediction $j_2$, $r \in \widehat{\mathfrak{A}}_{j_2} \setminus \{r_{g_2 \cdot y_1}\}$ cannot be activated, moreover, we have*

$$\Lambda^{(t)}_{5,j_2,r_{g_2 \cdot y_1}} = \left(\mathbf{Attn}^{(t)}_{\mathsf{ans},1\to\mathsf{pred},2} - \mathbf{Attn}^{(t)}_{\mathsf{ans},1\to\mathsf{ans},0}\right) \cdot 2B$$
$$+ \left(\mathbf{Attn}^{(t)}_{\mathsf{ans},1\to\mathsf{ans},1} - \mathbf{Attn}^{(t)}_{\mathsf{ans},1\to\mathsf{pred},1}\right) \cdot \frac{2B}{n_y} + O(\delta).$$

*2. for the prediction $j_2' = \tau\big(g_2(y_0)\big)$, $r \in \widehat{\mathfrak{A}}_{j_2'} \setminus \{r_{g_2 \cdot y_0}\}$ (notice that in this case $r_{g_2 \cdot y_0} = r_{g_1 \cdot y_0}$) cannot be activated, moreover, we have*

$$\Lambda^{(t)}_{5,j_2',r_{g_2 \cdot y_0}} = \left(\mathbf{Attn}^{(t)}_{\mathsf{ans},1\to\mathsf{pred},2} + \mathbf{Attn}^{(t)}_{\mathsf{ans},1\to\mathsf{pred},1} - \mathbf{Attn}^{(t)}_{\mathsf{ans},1\to\mathsf{ans},1}\right) \cdot 2B + O(\delta).$$

*3. for the prediction $\tau(g_1(y_1))$, $r \in \widehat{\mathfrak{A}}_{\tau(g_1(y_1))} \setminus \{r_{g_1 \cdot y_1}\}$ cannot be activated, moreover, we have*

$$\Lambda^{(t)}_{5,\tau(g_1(y_1)),r_{g_1 \cdot y_1}} = \left(\mathbf{Attn}^{(t)}_{\mathsf{ans},1\to\mathsf{ans},1} - \mathbf{Attn}^{(t)}_{\mathsf{ans},1\to\mathsf{pred},2}\right) \cdot \frac{2B}{n_y} + O(\delta).$$

4. *for other $j \in \tau(\mathcal{Y})$, if there are $j$ and $y$, s.t., $g_1, g_2 \in \mathrm{Fiber}_{j,y}$ (notice that $y \neq y_0, y_1$ for $\mathcal{E}_2$), then for such a $j$, $r \in \widehat{\mathfrak{A}}_j \setminus \{r_{g_2 \cdot y}\}$ cannot be activated, moreover, we have*

$$\Lambda_{5,j,r_{g_2 \cdot y}}^{(t)} = \left(\mathbf{Attn}_{\mathsf{ans},1\to\mathsf{pred},2}^{(t)} - \mathbf{Attn}_{\mathsf{ans},1\to\mathsf{ans},1}^{(t)}\right) \cdot 2B + \widetilde{O}\left(\frac{B}{d \cdot n_y}\right) + O(\delta);$$

*else, none of $r \in \widehat{\mathfrak{A}}_j$ can be activated.*

**Lemma H.15.** *If Induction H.1 holds for all iterations $< t$, then given $\mathbf{Z}^{2,1} \in \mathcal{E}_3$, we have*

1. *for the prediction $j_2$, $r \in \widehat{\mathfrak{A}}_{j_2} \setminus \{r_{g_2 \cdot y_1}\}$ cannot be activated, moreover, we have*

$$\Lambda_{5,j_2,r_{g_2 \cdot y_1}}^{(t)} = \mathbf{Attn}_{\mathsf{ans},1\to\mathsf{pred},2}^{(t)} \cdot 2B \pm O(\delta).$$

2. *for the prediction $\tau(g_1(y_1))$, $r \in \widehat{\mathfrak{A}}_{\tau(g_1(y_1))} \setminus \{r_{g_1 \cdot y_1}\}$ cannot be activated, moreover, we have*

$$\Lambda_{5,\tau(g_1(y_1)),r_{g_1 \cdot y_1}}^{(t)} = \mathbf{Attn}_{\mathsf{ans},1\to\mathsf{pred},1}^{(t)} \cdot 2B \pm O(\delta).$$

3. *for other $j = g(y_1)$, where $g_1, g_2 \notin \mathrm{Fiber}_{j,y_1}$, we have*

$$\Lambda_{5,j,r_g \cdot y_1}^{(t)} = \left(\mathbf{Attn}_{\mathsf{ans},1\to\mathsf{ans},1}^{(t)} - \mathbf{Attn}_{\mathsf{ans},1\to\mathsf{pred},2}^{(t)}\right) \cdot \frac{2B}{n_y} + \widetilde{O}\left(\frac{B}{d \cdot n_y}\right) + O(\delta);$$

*moreover, if there exists $y$, s.t., $g_1, g_2 \in \mathrm{Fiber}_{j,y}$ (notice that $y \neq y_1$ for $\mathcal{E}_3$), we have*

$$\Lambda_{5,j,r_{g_2 \cdot y}}^{(t)} = \left(\mathbf{Attn}_{\mathsf{ans},1\to\mathsf{pred},2}^{(t)} - \mathbf{Attn}_{\mathsf{ans},1\to\mathsf{ans},1}^{(t)}\right) \cdot 2B + \widetilde{O}\left(\frac{B}{d \cdot n_y}\right) + O(\delta);$$

*besides, other $r \in \widehat{\mathfrak{A}}_j$ cannot be activated.*

**Lemma H.16.** *If Induction H.1 holds for all iterations $< t$, then given $\mathbf{Z}^{2,1} \in \mathcal{E}_4$, we have*

1. *for the prediction $j_2$, $r \in \widehat{\mathfrak{A}}_{j_2} \setminus \{r_{g_2 \cdot y_1}\}$ cannot be activated, moreover, we have*

$$\Lambda_{5,j_2,r_{g_2 \cdot y_1}}^{(t)} = \mathbf{Attn}_{\mathsf{ans},1\to\mathsf{pred},2}^{(t)} \cdot 2B + O(\delta).$$

2. *for the prediction $\tau(g_2(y_0))$, $r \in \widehat{\mathfrak{A}}_{\tau(g_2(y_0))} \setminus \{r_{g_2 \cdot y_0}\}$ cannot be activated, moreover, we have*

$$\Lambda_{5,j_2',r_{g_2 \cdot y_0}}^{(t)} = \left(\mathbf{Attn}_{\mathsf{ans},1\to\mathsf{pred},2}^{(t)} + \mathbf{Attn}_{\mathsf{ans},1\to\mathsf{pred},1}^{(t)} - \mathbf{Attn}_{\mathsf{ans},1\to\mathsf{ans},1}^{(t)}\right) \cdot 2B + O(\delta).$$

3. *for other $j \in \tau(\mathcal{Y})$, if there exist $j$ and $y$, s.t., $g_1, g_2 \in \mathrm{Fiber}_{j,y}$, then for such a $j$, $r \in \widehat{\mathfrak{A}}_j \setminus \{r_{g_2 \cdot y}\}$ cannot be activated, moreover, we have*

$$\Lambda_{5,j,r_{g_2 \cdot y}}^{(t)} = \left(\mathbf{Attn}_{\mathsf{ans},1\to\mathsf{pred},2}^{(t)} - \mathbf{Attn}_{\mathsf{ans},1\to\mathsf{ans},1}^{(t)}\right) \cdot 2B + \widetilde{O}\left(\frac{B}{d \cdot n_y}\right) + O(\delta);$$

*else, none of $r \in \widehat{\mathfrak{A}}_j$ can be activated.*

**Lemma H.17.** *If Induction H.1 holds for all iterations $< t$, then given $\mathbf{Z}^{2,1} \in \mathcal{E}_5$, we have*

1. *for the prediction $j_2$, $r \in \widehat{\mathfrak{A}}_{j_2} \setminus \{r_{g_2 \cdot y_1}\}$ cannot be activated, moreover, we have*

$$\Lambda_{5,j_2,r_{g_2 \cdot y_1}}^{(t)} = \mathbf{Attn}_{\mathsf{ans},1\to\mathsf{pred},2}^{(t)} \cdot 2B \pm O(\delta).$$

2. *for the prediction $\tau(g_2(y_0))$, $r \in \widehat{\mathfrak{A}}_{\tau(g_2(y_0))} \setminus \{r_{g_2 \cdot y_0}\}$ cannot be activated, moreover, we have*

$$\Lambda_{5,j_2',r_{g_2 \cdot y_0}}^{(t)} = \left(\mathbf{Attn}_{\mathsf{ans},1\to\mathsf{pred},2}^{(t)} - \mathbf{Attn}_{\mathsf{ans},1\to\mathsf{ans},1}^{(t)}\right) \cdot 2B + \widetilde{O}\left(\frac{B}{d \cdot n_y}\right) + O(\delta).$$

3. *for the prediction $\tau(g_1(y_0))$, $r \in \widehat{\mathfrak{A}}_{\tau(g_2(y_0))} \setminus \{r_{g_1 \cdot y_0}\}$ cannot be activated, moreover, we have*

$$\Lambda^{(t)}_{5,\tau(g_1(y_0)),r_{g_1 \cdot y_0}} = \left(\mathbf{Attn}^{(t)}_{\mathsf{ans},1\to\mathsf{pred},1} - \mathbf{Attn}^{(t)}_{\mathsf{ans},1\to\mathsf{ans},1}\right) \cdot 2B$$
$$+ \left(\mathbf{Attn}^{(t)}_{\mathsf{ans},1\to\mathsf{ans},0} - \mathbf{Attn}^{(t)}_{\mathsf{ans},1\to\mathsf{pred},2}\right) \cdot \frac{2B}{n_y} + O(\delta).$$

4. *for other $j \in \tau(\mathcal{Y})$, if there are $j$ and $y$, s.t., $g_1, g_2 \in \mathrm{Fiber}_{j,y}$, then for such a $j$, $r \in \widehat{\mathfrak{A}}_j \setminus \{r_{g_2 \cdot y}\}$ cannot be activated, moreover, we have*

$$\Lambda^{(t)}_{5,j,r_{g_2 \cdot y}} = \left(\mathbf{Attn}^{(t)}_{\mathsf{ans},1\to\mathsf{pred},2} - \mathbf{Attn}^{(t)}_{\mathsf{ans},1\to\mathsf{ans},1}\right) \cdot 2B + \widetilde{O}\left(\frac{B}{d \cdot n_y}\right) + O(\delta);$$

*else, none of $r \in \widehat{\mathfrak{A}}_j$ can be activated.*

**Lemma H.18.** *If Induction H.1 holds for all iterations $< t$, then given $\mathbf{Z}^{2,1} \in \mathcal{E}_6$, we have*

1. *for the prediction $j_2$, $r \in \widehat{\mathfrak{A}}_{j_2} \setminus \{r_{g_2 \cdot y_1}\}$ cannot be activated, moreover, we have*

$$\Lambda^{(t)}_{5,j_2,r_{g_2 \cdot y_1}} = \left(\mathbf{Attn}^{(t)}_{\mathsf{ans},1\to\mathsf{pred},2} + \mathbf{Attn}^{(t)}_{\mathsf{ans},1\to\mathsf{pred},1}\right) \cdot 2B \pm O(\delta).$$

2. *for other $j = g(y_1)$, where $g_2 \notin \mathrm{Fiber}_{j,y_1}$, we have*

$$\Lambda^{(t)}_{5,j,r_{g \cdot y_1}} = \left(\mathbf{Attn}^{(t)}_{\mathsf{ans},1\to\mathsf{ans},1} - \mathbf{Attn}^{(t)}_{\mathsf{ans},1\to\mathsf{pred},2}\right) \cdot \frac{2B}{n_y} + \widetilde{O}\left(\frac{B}{d \cdot n_y}\right) + O(\delta);$$

*moreover, if there exists $y \neq y_1$, s.t., $g_1, g_2 \in \mathrm{Fiber}_{j,y}$, we have*

$$\Lambda^{(t)}_{5,j,r_{g_2 \cdot y}} = \left(\mathbf{Attn}^{(t)}_{\mathsf{ans},1\to\mathsf{pred},2} - \mathbf{Attn}^{(t)}_{\mathsf{ans},1\to\mathsf{ans},1}\right) \cdot 2B + \widetilde{O}\left(\frac{B}{d \cdot n_y}\right) + O(\delta);$$

*besides, other $r \in \widehat{\mathfrak{A}}_j$ cannot be activated.*

### H.3.2 Gradient Lemma

**Lemma H.19.** *If Induction H.1 holds for all iterations $< t$, given $s \in \tau(\mathcal{X})$, we have*

$$\left[-\nabla_{\mathbf{Q}^{(t)}_{4,3}} \mathsf{Loss}^{2,2}_5\right]_{s,s} + \left[-\nabla_{\mathbf{Q}^{(t)}_{4,4}} \mathsf{Loss}^{2,2}_5\right]_{s,s}$$
$$\geq \min\left\{\Omega\left(\frac{1}{d \cdot n_y}\right), \Omega\left(\frac{B}{d}\right) \cdot \mathbb{E}\left[(1 - \mathbf{logit}^{(t)}_{5,j_2})|\tau(x_1) = s, \mathcal{E}_1\right]\right\} > 0.$$

*Proof.* By Lemma H.10 and Lemma H.11, we have $\left[-\nabla_{\mathbf{Q}_{4,3}} \mathsf{Loss}^{2,2}_5\right]_{s,s} + \left[-\nabla_{\mathbf{Q}_{4,4}} \mathsf{Loss}^{2,2}_5\right]_{s,s} = \mathbb{E}\left[\mathcal{N}_{s,2,i} + \mathcal{N}_{s,2,ii} + \mathcal{N}_{s,2,iii}\right]$. Based onLemma H.13-Lemma H.18, we can first directly bound the term $\mathcal{N}^{(t)}_{s,2,iii}$ as follows:

$$\mathbb{E}\left[\mathcal{N}^{(t)}_{s,2,iii}\right] \leq \widetilde{O}(\sigma_0^q) = \frac{1}{\mathsf{poly}d}.$$

In the following discussion, we focus on $\mathcal{N}^{(t)}_{s,2,i}$ and $\mathcal{N}^{(t)}_{s,2,ii}$, and consider two regimes for the gradient lower bound: (i) $\mathbb{E}\left[\mathbf{Attn}^{(t)}_{\mathsf{ans},1\to\mathsf{pred},2} - \mathbf{Attn}^{(t)}_{\mathsf{ans},1\to\mathsf{ans},0} \mid \tau(x_1) = s\right] \leq \frac{\varrho}{B}$, and (ii) $\mathbb{E}\left[\mathbf{Attn}^{(t)}_{\mathsf{ans},1\to\mathsf{pred},2} - \mathbf{Attn}^{(t)}_{\mathsf{ans},1\to\mathsf{ans},0} \mid \tau(x_1) = s\right] \geq \frac{\varrho}{B}$. The proof strategy is to analyze $\mathcal{N}^{(t)}_{s,2,i}$ and $\mathcal{N}^{(t)}_{s,2,ii}$ under different event conditions in two regimes.

1. For regime (i),

(a) For $\mathbf{Z}^{2,1} \in \mathcal{E}_1$, by Induction H.1 and Lemma H.13, we have an immediate logit upper bound for $j \in \tau(\mathcal{Y})$: $\mathbf{logit}_{5,j}^{(t)} \leq O(\frac{1}{d})$. Then by Lemma H.10, we have

$$\mathbb{E}\left[\mathcal{N}_{s,2,i}^{(t)} \mathbb{1}_{\mathcal{E}_1}\right]$$

$$= \mathbb{E}\left[(1 - \mathbf{logit}_{5,j_2}^{(t)}) \cdot \Big(\sum_{r \in \widehat{\mathfrak{A}}_{j_2}} \mathbf{sReLU}'(\Lambda_{5,j_2,r}^{(t)})\cdot\right.$$

$$\left(-\mathbf{Attn}_{\mathsf{ans},1\to\mathsf{ans},0}^{(t)} \cdot V_{j_2,r}(y_0) - \mathbf{Attn}_{\mathsf{ans},1\to\mathsf{pred},1}^{(t)} \cdot V_{j_2,r}(g_1)\right.$$

$$\left.+ \left(1 - \mathbf{Attn}_{\mathsf{ans},1\to\mathsf{ans},1}^{(t)} - \mathbf{Attn}_{\mathsf{ans},1\to\mathsf{pred},2}^{(t)}\right)\Lambda_{5,j_2,r}^{(t)} \pm \widetilde{O}(\sigma_0)\Big) \pm \widetilde{O}(\delta^q)\right) \mathbb{1}_{\tau(x_1)=s}\mathbb{1}_{\mathcal{E}_1}\Big].$$

By Lemma H.13, we can obtain

$$\mathbb{E}\left[(1 - \mathbf{logit}_{5,j_2}^{(t)}) \cdot \Big(\mathbf{sReLU}'(\Lambda_{5,j_2,r_{g_2\cdot y_1}}^{(t)})\cdot\right.$$

$$\left(-\mathbf{Attn}_{\mathsf{ans},1\to\mathsf{ans},0}^{(t)} \cdot V_{j_2,r_{g_2\cdot y_1}}(y_0) - \mathbf{Attn}_{\mathsf{ans},1\to\mathsf{pred},1}^{(t)} \cdot V_{j_2,r_{g_2\cdot y_1}}(g_1)\right.$$

$$\left.+ \left(1 - \mathbf{Attn}_{\mathsf{ans},1\to\mathsf{ans},1}^{(t)} - \mathbf{Attn}_{\mathsf{ans},1\to\mathsf{pred},2}^{(t)}\right)\Lambda_{5,j_2,r_{g_2\cdot y_1}}^{(t)} \pm \widetilde{O}(\sigma_0)\Big) \pm \widetilde{O}(\delta^q)\right) \mathbb{1}_{\tau(x_1)=s}\mathbb{1}_{\mathcal{E}_1}\Big]$$

$$\geq \Omega\Big(\frac{B}{d}\Big) \cdot \mathbb{E}\left[\mathbf{sReLU}'(\Lambda_{5,j_2,r_{g_2\cdot y_1}}^{(t)})\big|\tau(x_1) = s, \mathcal{E}_1\right]. \tag{109}$$

On the other hand,

$$-\mathbb{E}\left[(1 - \mathbf{logit}_{5,j_2}^{(t)}) \cdot \mathbf{sReLU}'(\Lambda_{5,j_2,r_{g_1\cdot y_0}}^{(t)})\cdot\right.$$

$$\left(\mathbf{Attn}_{\mathsf{ans},1\to\mathsf{ans},0}^{(t)} \cdot V_{j_2,r_{g_1\cdot y_0}}(y_0) + \mathbf{Attn}_{\mathsf{ans},1\to\mathsf{pred},1}^{(t)} \cdot V_{j_2,r_{g_1\cdot y_0}}(g_1)\right)$$

$$\left.\mathbb{1}_{\tau(x_1)=s}\mathbb{1}_{\mathcal{E}_1}\mathbb{1}_{g_1(y_0)=g_2(y_1)}\right]$$

$$\geq -O\Big(\frac{B}{d \cdot n_y}\Big) \cdot \mathbb{E}\left[\mathbf{sReLU}'(\Lambda_{5,j_2,r_{g_2\cdot y_1}}^{(t)})\big|\tau(x_1) = s, \mathcal{E}_1\right],$$

which implies that $\mathbb{E}\left[\mathcal{N}_{s,2,i}^{(t)} \mathbb{1}_{\mathcal{E}_1}\right] \geq \Omega\Big(\frac{B}{d}\Big) \cdot \mathbb{E}\left[\mathbf{sReLU}'(\Lambda_{5,j_2,r_{g_2\cdot y_1}}^{(t)})\big|\tau(x_1) = s, \mathcal{E}_1\right] \geq 0$. Moving to $\mathcal{N}_{s,2,ii}^{(t)}$, by Lemma H.10 and Lemma H.13, we have

$$\mathbb{E}\left[\mathcal{N}_{s,2,ii}^{(t)} \mathbb{1}_{\mathcal{E}_1}\right]$$

$$= \mathbb{E}\left[\sum_{j\neq j_2 \in \tau(\mathcal{Y})} \mathbf{logit}_{5,j}^{(t)} \cdot \Big(\sum_{r \in \widehat{\mathfrak{A}}_j} \mathbf{sReLU}'(\Lambda_{5,j,r})\cdot\right.$$

$$\left(\mathbf{Attn}_{\mathsf{ans},1\to\mathsf{ans},0}^{(t)} \cdot V_{j,r}(y_0) + \mathbf{Attn}_{\mathsf{ans},1\to\mathsf{pred},1}^{(t)} \cdot V_{j,r}(g_1)\right.$$

$$\left.- \left(1 - \mathbf{Attn}_{\mathsf{ans},1\to\mathsf{ans},1}^{(t)} - \mathbf{Attn}_{\mathsf{ans},1\to\mathsf{pred},2}^{(t)}\right)\Lambda_{5,j,r}^{(t)} \pm \widetilde{O}(\sigma_0)\Big) \pm \widetilde{O}(\delta^q)\right) \mathbb{1}_{\tau(x_1)=s}\mathbb{1}_{\mathcal{E}_1}\Big]$$

$$\geq -O\Big(\frac{n_y \cdot B}{d^2}\Big).$$

(b) For $\mathbf{Z}^{2,1} \in \mathcal{E}_2$, by Induction H.1 and Lemma H.14, we can also obtain a crude bound of logit: $\mathbf{logit}_{5,j}^{(t)} \leq O(\frac{1}{d})$ for $j \neq j_2'$. Then $\mathcal{N}_{s,2,i}^{(t)}$ can be bounded in the same way as

 and we have

$$\mathbb{E}\left[\mathcal{N}_{s,2,i}^{(t)}\mathbb{1}_{\mathcal{E}_2}\right] \geq \Omega\left(\frac{B}{d \cdot n_y}\right) \cdot \mathbb{E}\left[\mathbf{sReLU}'(\Lambda_{5,j_2,r_{g_2 \cdot y_1}}^{(t)})\Big|\tau(x_1)=s,\mathcal{E}_2\right] \geq 0.$$

Moving to $\mathcal{N}_{s,2,ii}^{(t)}$,

• for $j = \tau(g_1(y_1))$, we have

$$= \mathbb{E}\bigg[\mathbf{logit}_{5,j}^{(t)}\Big(\mathbf{sReLU}'(\Lambda_{5,j,r_{g_1 \cdot y_1}})\cdot$$

$$\Big(\mathbf{Attn}_{\mathsf{ans},1\to\mathsf{ans},0}^{(t)} \cdot V_{j,r_{g_1 \cdot y_1}}(y_0) + \mathbf{Attn}_{\mathsf{ans},1\to\mathsf{pred},1}^{(t)} \cdot V_{j,r_{g_1 \cdot y_1}}(g_1)$$

$$- \big(1 - \mathbf{Attn}_{\mathsf{ans},1\to\mathsf{ans},1}^{(t)} - \mathbf{Attn}_{\mathsf{ans},1\to\mathsf{pred},2}^{(t)}\big)\Lambda_{5,j,r}^{(t)} \pm \widetilde{O}(\sigma_0)\Big) \pm \widetilde{O}(\delta^q)\Big)\mathbb{1}_{\tau(x_1)=s}\mathbb{1}_{\mathcal{E}_2}\bigg]$$

$$\geq -O\left(\frac{B}{n_y^2 \cdot d^2}\right).$$

where the last inequality is due to the cancellation of the term $\mathbf{Attn}_{\mathsf{ans},1\to\mathsf{ans},0}^{(t)} \cdot V_{j,r_{g_1 \cdot y_1}}(y_0) + \mathbf{Attn}_{\mathsf{ans},1\to\mathsf{pred},1}^{(t)} \cdot V_{j,r_{g_1 \cdot y_1}}(g_1)$ and the fact that

$$\Lambda_{5,\tau(g_1(y_1)),r_{g_1 \cdot y_1}}^{(t)} = \Big(\mathbf{Attn}_{\mathsf{ans},1\to\mathsf{ans},1}^{(t)} - \mathbf{Attn}_{\mathsf{ans},1\to\mathsf{pred},2}^{(t)}\Big) \cdot \frac{2B}{n_y} + O(\delta) \leq O\left(\frac{B}{n_y}\right).$$

• for $j = \tau(g_2(y))$ if there exists $y \neq y_0, y_1$ s.t., $g_1(y) = g_2(y)$

  – when $\mathbb{E}\big[\mathbf{Attn}_{\mathsf{ans},1\to\mathsf{ans},1}^{(t)} - \mathbf{Attn}_{\mathsf{ans},1\to\mathsf{pred},2}^{(t)} \mid \tau(x_1)=s\big] \leq 0.01$, by Induction H.1 and Lemma H.14, we have $\mathbf{logit}_{5,\tau(g_2(y))}^{(t)} \ll O\left(\frac{1}{d}\right) \cdot \mathbf{logit}_{5,j_2'}^{(t)}$ and $\mathbf{logit}_{5,j_2'}^{(t)} = \Omega(1)$. Then

$$\left|\mathbb{E}\bigg[\mathbf{logit}_{5,j}^{(t)} \cdot \Big(\mathbf{sReLU}'(\Lambda_{5,j,r_{g_1 \cdot y}}^{(t)})\cdot\right.$$

$$\Big(\mathbf{Attn}_{\mathsf{ans},1\to\mathsf{ans},0}^{(t)} \cdot V_{j,r_{g_1 \cdot y}}(y_0) + \mathbf{Attn}_{\mathsf{ans},1\to\mathsf{pred},1}^{(t)} \cdot V_{j,r_{g_1 \cdot y}}(g_1)$$

$$- \big(1 - \mathbf{Attn}_{\mathsf{ans},1\to\mathsf{ans},1}^{(t)} - \mathbf{Attn}_{\mathsf{ans},1\to\mathsf{pred},2}^{(t)}\big)\Lambda_{5,j,r_{g_1 \cdot y}}^{(t)} \pm \widetilde{O}(\sigma_0)\Big) \pm \widetilde{O}(\delta^q)\Big)\mathbb{1}_{\tau(x_1)=s}\mathbb{1}_{\mathcal{E}_2}\bigg]\right|$$

$$\leq O\left(\frac{B}{n_y \cdot d^2}\right).$$

On the other hand,

$$\mathbb{E}\bigg[\mathbf{logit}_{5,j_2'}^{(t)} \cdot \Big(\mathbf{sReLU}'(\Lambda_{5,j_2',r_{g_1 \cdot y_0}}^{(t)})\cdot$$

$$\Big(\mathbf{Attn}_{\mathsf{ans},1\to\mathsf{ans},0}^{(t)} \cdot V_{j_2',r_{g_1 \cdot y_0}}(y_0) + \mathbf{Attn}_{\mathsf{ans},1\to\mathsf{pred},1}^{(t)} \cdot V_{j_2',r_{g_1 \cdot y_0}}(g_1)$$

$$- \big(1 - \mathbf{Attn}_{\mathsf{ans},1\to\mathsf{ans},1}^{(t)} - \mathbf{Attn}_{\mathsf{ans},1\to\mathsf{pred},2}^{(t)}\big)\Lambda_{5,j_2',r_{g_1 \cdot y_0}}^{(t)} \pm \widetilde{O}(\sigma_0)\Big)$$

$$\pm \widetilde{O}(\delta^q)\Big)\mathbb{1}_{\tau(x_1)=s}\mathbb{1}_{\mathcal{E}_1}\bigg]$$

$$\geq \Omega\left(\frac{B}{n_y \cdot d}\right),$$

where the last inequality follows from the fact that

$$\mathbf{Attn}_{\mathsf{ans},1\to\mathsf{ans},0}^{(t)} \cdot V_{j_2',r_{g_1 \cdot y_0}}(y_0) + \mathbf{Attn}_{\mathsf{ans},1\to\mathsf{pred},1}^{(t)} \cdot V_{j_2',r_{g_1 \cdot y_0}}(g_1)$$

$$- \big(1 - \mathbf{Attn}_{\mathsf{ans},1\to\mathsf{ans},1}^{(t)} - \mathbf{Attn}_{\mathsf{ans},1\to\mathsf{pred},2}^{(t)}\big)\Lambda_{5,j_2',r_{g_1 \cdot y_0}}^{(t)}$$

$$= \mathbf{Attn}^{(t)}_{\mathsf{ans},1\to\mathsf{pred},1} \cdot \left(2B - 2\Lambda^{(t)}_{5,j_2',r_{g_1\cdot y_0}}\right) \pm O(\delta) \geq \Omega(B).$$

– when $\mathbb{E}\left[\mathbf{Attn}^{(t)}_{\mathsf{ans},1\to\mathsf{ans},1} - \mathbf{Attn}^{(t)}_{\mathsf{ans},1\to\mathsf{pred},2} \mid \tau(x_1) = s\right] \geq 0.01$, by Lemma H.14, we have $\Lambda^{(t)}_{5,j,r_{g_1\cdot y}}$ cannot be activated. Furthermore,

$$\mathbb{E}\Bigg[\mathbf{logit}^{(t)}_{5,j_2'} \cdot \Bigg(\mathbf{sReLU}'(\Lambda^{(t)}_{5,j_2',r_{g_1\cdot y_0}})\cdot$$
$$\left(\mathbf{Attn}^{(t)}_{\mathsf{ans},1\to\mathsf{ans},0} \cdot V_{j_2',r_{g_1\cdot y_0}}(y_0) + \mathbf{Attn}^{(t)}_{\mathsf{ans},1\to\mathsf{pred},1} \cdot V_{j_2',r_{g_1\cdot y_0}}(g_1)\right.$$
$$\left.- \left(1 - \mathbf{Attn}^{(t)}_{\mathsf{ans},1\to\mathsf{ans},1} - \mathbf{Attn}^{(t)}_{\mathsf{ans},1\to\mathsf{pred},2}\right)\Lambda^{(t)}_{5,j_2',r_{g_1\cdot y_0}} \pm \widetilde{O}(\sigma_0)\right) \pm \widetilde{O}(\delta^q)\Bigg)\mathbb{1}_{\tau(x_1)=s}\mathbb{1}_{\mathcal{E}_1}\Bigg]$$
$$\geq 0.$$

Putting the above discussion together, we have

$$\mathbb{E}\left[\mathcal{N}^{(t)}_{s,2,ii}\mathbb{1}_{\mathcal{E}_2}\right] \geq -O\left(\frac{B}{n_y^2 \cdot d^2}\right).$$

(c) For $\mathbf{Z}^{2,1} \in \mathcal{E}_3$, by Induction H.1 and Lemma H.15, we can first have some facts of logit:

$$1 - \mathbf{logit}^{(t)}_{5,j_2} = \Omega(1), \quad \mathbf{logit}^{(t)}_{5,\tau(g_1(y_1))} = \Omega(1)$$
$$\mathbf{logit}^{(t)}_{5,j} \leq \frac{1}{\mathsf{poly}d} \text{ for other } j.$$

Therefore, we have

$$\mathbb{E}\left[\mathcal{N}^{(t)}_{s,2,i}\mathbb{1}_{\mathcal{E}_3}\right]$$
$$= \mathbb{E}\Bigg[(1 - \mathbf{logit}^{(t)}_{5,j_2}) \cdot \Bigg(- \mathbf{Attn}^{(t)}_{\mathsf{ans},1\to\mathsf{ans},0} \cdot V_{j_2,r_{g_2\cdot y_1}}(y_1) - \mathbf{Attn}^{(t)}_{\mathsf{ans},1\to\mathsf{pred},1} \cdot V_{j_2,r_{g_2\cdot y_1}}(g_1)$$
$$+ \left(1 - \mathbf{Attn}^{(t)}_{\mathsf{ans},1\to\mathsf{ans},1} - \mathbf{Attn}^{(t)}_{\mathsf{ans},1\to\mathsf{pred},2}\right)\Lambda^{(t)}_{5,j_2,r_{g_2\cdot y_1}} \pm \widetilde{O}(\sigma_0) \pm \widetilde{O}(\delta^q)\Bigg)\mathbb{1}_{\tau(x_1)=s}\mathbb{1}_{\mathcal{E}_3}\Bigg]$$
$$\geq \Omega\left(\frac{B}{n_y \cdot d}\right).$$

Moving to $\mathcal{N}^{(t)}_{s,2,ii}$, we have

$$\mathbb{E}\left[\mathcal{N}^{(t)}_{s,2,ii}\mathbb{1}_{\mathcal{E}_3}\right]$$
$$\geq \mathbb{E}\Bigg[\mathbf{logit}^{(t)}_{5,\tau(g_1(y_1))}\cdot$$
$$\left(\mathbf{Attn}^{(t)}_{\mathsf{ans},1\to\mathsf{ans},0} \cdot V_{\tau(g_1(y_1)),r_{g_1\cdot y_1}}(y_1) + \mathbf{Attn}^{(t)}_{\mathsf{ans},1\to\mathsf{pred},1} \cdot V_{\tau(g_1(y_1)),r}(g_1)\right.$$
$$\left.- \left(1 - \mathbf{Attn}^{(t)}_{\mathsf{ans},1\to\mathsf{ans},1} - \mathbf{Attn}^{(t)}_{\mathsf{ans},1\to\mathsf{pred},2}\right)\Lambda^{(t)}_{5,\tau(g_1(y_1)),r_{g_1\cdot y_1}} \pm \widetilde{O}(\sigma_0)\right.$$
$$\left.\pm \widetilde{O}(\delta^q)\right)\mathbb{1}_{\tau(x_1)=s}\mathbb{1}_{\mathcal{E}_3}\Bigg] - O\left(\frac{n_y \cdot B}{d \cdot \mathsf{poly}d}\right)$$
$$\geq \Omega\left(\frac{B}{n_y \cdot d}\right) - O\left(\frac{B}{d \cdot \mathsf{poly}d}\right) = \Omega\left(\frac{B}{n_y \cdot d}\right).$$

(d) For $\mathbf{Z}^{2,1} \in \mathcal{E}_4$, by comparing Lemma H.16 with Lemma H.14, we can directly bound $\mathcal{N}^{(t)}_{s,2,ii}$ in the same way as $\mathcal{E}_2$, where the only difference is that we do not need to

consider $\tau(g_1(y_1))$ for $\mathcal{E}_4$. Thus $\mathbb{E}\left[\mathcal{N}_{s,2,ii}^{(t)} \mathbb{1}_{\mathcal{E}_4}\right] \geq 0$. Moreover, it is clear that

$$\mathbb{E}\left[\mathcal{N}_{s,2,i}^{(t)} \mathbb{1}_{\mathcal{E}_4}\right]$$

$$= \mathbb{E}\Bigg[(1 - \mathbf{logit}_{5,j_2}^{(t)}) \cdot \Big(\mathbf{sReLU}'(\Lambda_{5,j_2,r_{g_2 \cdot y_1}}^{(t)})\cdot$$

$$\Big(- \mathbf{Attn}_{\mathsf{ans},1\to\mathsf{ans},0}^{(t)} \cdot V_{j_2,r_{g_2 \cdot y_1}}(y_0) - \mathbf{Attn}_{\mathsf{ans},1\to\mathsf{pred},1}^{(t)} \cdot V_{j_2,r_{g_2 \cdot y_1}}(g_1)$$

$$+ \big(1 - \mathbf{Attn}_{\mathsf{ans},1\to\mathsf{ans},1}^{(t)} - \mathbf{Attn}_{\mathsf{ans},1\to\mathsf{pred},2}^{(t)}\big)\Lambda_{5,j_2,r_{g_2 \cdot y_1}}^{(t)} \pm \widetilde{O}(\sigma_0)\Big) \pm \widetilde{O}(\delta^q)\Big)\mathbb{1}_{\tau(x_1)=s}\mathbb{1}_{\mathcal{E}_4}\Bigg]$$

$$\geq 0.$$

where the last inequality is due to the cancellation of $\mathbf{Attn}_{\mathsf{ans},1\to\mathsf{ans},0}^{(t)} \cdot V_{j_2,r_{g_2 \cdot y_1}}(y_0) + \mathbf{Attn}_{\mathsf{ans},1\to\mathsf{pred},1}^{(t)} \cdot V_{j_2,r_{g_2 \cdot y_1}}(g_1)$, and the fact that

$$\Lambda_{5,j_2,r_{g_2 \cdot y_1}}^{(t)} = \mathbf{Attn}_{\mathsf{ans},1\to\mathsf{pred},2}^{(t)} \cdot 2B + O(\delta) \geq \Omega(B).$$

(e) For $\mathbf{Z}^{2,1} \in \mathcal{E}_5 \cup \mathcal{E}_6$, by Induction H.1, Lemma H.17 and Lemma H.18, we can derive the following logit condition: $1 - \mathbf{logit}_{5,j_2}^{(t)}, \mathbf{logit}_{5,j}^{(t)} \leq \frac{1}{\mathsf{poly}d}$ for $j \neq j_2$. Hence, we can simply bound $\mathcal{N}_{s,2,i}^{(t)}$ and $\mathcal{N}_{s,2,ii}^{(t)}$ as follows:

$$\left|\mathbb{E}\left[\mathcal{N}_{s,2,i}^{(t)} \mathbb{1}_{\mathcal{E}_m}\right]\right|, \left|\mathbb{E}\left[\mathcal{N}_{s,2,ii}^{(t)} \mathbb{1}_{\mathcal{E}_m}\right]\right| \leq O\Big(\frac{B}{dn_y} \cdot \frac{1}{\mathsf{poly}d}\Big) \text{ for } m \in \{5,6\}.$$

Putting it all together, we have for the regmie (1),

$$\left[-\nabla_{\mathbf{Q}_{4,3}^{(t)}} \mathsf{Loss}_5^{2,2}\right]_{s,s} + \left[-\nabla_{\mathbf{Q}_{4,4}^{(t)}} \mathsf{Loss}_5^{2,2}\right]_{s,s} = \sum_{\kappa \in \{i,ii,iii\}} \sum_{i \in [6]} \mathbb{E}\left[\mathcal{N}_{s,2,\kappa}^{(t)} \mathbb{1}_{\mathcal{E}_i}\right] \geq \Omega(\frac{B}{d \cdot n_y}).$$

2. In regime (ii), the analysis follows a structure analogous to that of regime (i). The primary distinction lies in the fact that, under the main event $\mathcal{E}_1$, the term $\Lambda_{5,j_2,r_{g_2 \cdot y_1}}^{(t)}$ is guaranteed to remain within the linear regime, thereby serving as the dominant component driving the overall gradient.

(a) For $\mathbf{Z}^{2,1} \in \mathcal{E}_1$, by Induction H.1 and Lemma H.13, we have

$$\mathbb{E}\left[\mathcal{N}_{s,2,i}^{(t)} \mathbb{1}_{\mathcal{E}_1}\right]$$

$$= \mathbb{E}\Bigg[(1 - \mathbf{logit}_{5,j_2}^{(t)}) \cdot \Big(\sum_{r \in \widehat{\mathfrak{A}}_{j_2}} \mathbf{sReLU}'(\Lambda_{5,j_2,r}^{(t)})\cdot$$

$$\Big(- \mathbf{Attn}_{\mathsf{ans},1\to\mathsf{ans},0}^{(t)} \cdot V_{j_2,r}(y_0) - \mathbf{Attn}_{\mathsf{ans},1\to\mathsf{pred},1}^{(t)} \cdot V_{j_2,r}(g_1)$$

$$+ \big(1 - \mathbf{Attn}_{\mathsf{ans},1\to\mathsf{ans},1}^{(t)} - \mathbf{Attn}_{\mathsf{ans},1\to\mathsf{pred},2}^{(t)}\big)\Lambda_{5,j_2,r}^{(t)} \pm \widetilde{O}(\sigma_0)\Big) \pm \widetilde{O}(\delta^q)\Big)\mathbb{1}_{\tau(x_1)=s}\mathbb{1}_{\mathcal{E}_1}\Bigg].$$

By Lemma H.13, we can obtain

$$\mathbb{E}\Bigg[(1 - \mathbf{logit}_{5,j_2}^{(t)}) \cdot \Big(\big(- \mathbf{Attn}_{\mathsf{ans},1\to\mathsf{ans},0}^{(t)} \cdot V_{j_2,r_{g_2 \cdot y_1}}(y_0) - \mathbf{Attn}_{\mathsf{ans},1\to\mathsf{pred},1}^{(t)} \cdot V_{j_2,r_{g_2 \cdot y_1}}(g_1)$$

$$+ \big(1 - \mathbf{Attn}_{\mathsf{ans},1\to\mathsf{ans},1}^{(t)} - \mathbf{Attn}_{\mathsf{ans},1\to\mathsf{pred},2}^{(t)}\big)\Lambda_{5,j_2,r_{g_2 \cdot y_1}}^{(t)} \pm \widetilde{O}(\sigma_0)\Big) \pm \widetilde{O}(\delta^q)\Big)\mathbb{1}_{\tau(x_1)=s}\mathbb{1}_{\mathcal{E}_1}\Bigg]$$

$$\tag{110}$$

$$\geq \Omega\Big(\frac{B}{d}\Big) \cdot \mathbb{E}\left[(1 - \mathbf{logit}_{5,j_2}^{(t)})\big|\tau(x_1) = s, \mathcal{E}_1\right].$$

On the other hand,

$$- \mathbb{E}\Bigg[(1 - \mathbf{logit}_{5,j_2}^{(t)}) \cdot \mathbf{sReLU}'(\Lambda_{5,j_2,r_{g_1 \cdot y_0}}^{(t)}) \cdot$$

$$\Big(\mathbf{Attn}_{\mathsf{ans},1 \to \mathsf{ans},0}^{(t)} \cdot V_{j_2,r_{g_1 \cdot y_0}}(y_0) + \mathbf{Attn}_{\mathsf{ans},1 \to \mathsf{pred},1}^{(t)} \cdot V_{j_2,r_{g_1 \cdot y_0}}(g_1)\Big)$$

$$\mathbb{1}_{\tau(x_1)=s} \mathbb{1}_{\mathcal{E}_1} \mathbb{1}_{g_1(y_0)=g_2(y_1)}\Bigg]$$

$$\geq -O\Big(\frac{B}{d \cdot n_y}\Big) \cdot \mathbb{E}\Big[(1 - \mathbf{logit}_{5,j_2}^{(t)})\big|\tau(x_1) = s, \mathcal{E}_1\Big],$$

which implies that $\mathbb{E}\Big[\mathcal{N}_{s,2,i}^{(t)} \mathbb{1}_{\mathcal{E}_1}\Big] \geq \Omega\Big(\frac{B}{d}\Big) \cdot \mathbb{E}\Big[(1 - \mathbf{logit}_{5,j_2}^{(t)})\big|\tau(x_1) = s, \mathcal{E}_1\Big] \geq 0.$

Moving to $\mathcal{N}_{s,2,ii}^{(t)}$, by Lemma H.13, we have

- for $j = \tau(g_1(y_0))$, $r = r_{g_1 \cdot y_0}$

$$\Big(\mathbf{Attn}_{\mathsf{ans},1 \to \mathsf{ans},0}^{(t)} \cdot V_{j,r}(y_0) + \mathbf{Attn}_{\mathsf{ans},1 \to \mathsf{pred},1}^{(t)} \cdot V_{j,r}(g_1)$$

$$- \big(1 - \mathbf{Attn}_{\mathsf{ans},1 \to \mathsf{ans},1}^{(t)} - \mathbf{Attn}_{\mathsf{ans},1 \to \mathsf{pred},2}^{(t)}\big)\Lambda_{5,j,r}^{(t)} \pm \widetilde{O}(\sigma_0)\Big) \geq 0.$$

- for $j = \tau(g_1(y_1))$, $r = r_{g_1 \cdot y_1}$, due to the cancellation of $\mathbf{Attn}_{\mathsf{ans},1 \to \mathsf{ans},0}^{(t)} \cdot V_{j,r}(y_0) + \mathbf{Attn}_{\mathsf{ans},1 \to \mathsf{pred},1}^{(t)} \cdot V_{j,r}(g_1)$, we have

$$\Big(\mathbf{Attn}_{\mathsf{ans},1 \to \mathsf{ans},0}^{(t)} \cdot V_{j,r}(y_0) + \mathbf{Attn}_{\mathsf{ans},1 \to \mathsf{pred},1}^{(t)} \cdot V_{j,r}(g_1)$$

$$- \big(1 - \mathbf{Attn}_{\mathsf{ans},1 \to \mathsf{ans},1}^{(t)} - \mathbf{Attn}_{\mathsf{ans},1 \to \mathsf{pred},2}^{(t)}\big)\Lambda_{5,j,r}^{(t)} \pm \widetilde{O}(\sigma_0)\Big)$$

$$\geq -\big(1 - \mathbf{Attn}_{\mathsf{ans},1 \to \mathsf{ans},1}^{(t)} - \mathbf{Attn}_{\mathsf{ans},1 \to \mathsf{pred},2}^{(t)}\big)\Lambda_{5,j,r}^{(t)}.$$

- for $j = \tau(g_2(y_0))$, $r = r_{g_2 \cdot y_0}$

$$\Big(\mathbf{Attn}_{\mathsf{ans},1 \to \mathsf{ans},0}^{(t)} \cdot V_{j,r}(y_0) + \mathbf{Attn}_{\mathsf{ans},1 \to \mathsf{pred},1}^{(t)} \cdot V_{j,r}(g_1)$$

$$- \big(1 - \mathbf{Attn}_{\mathsf{ans},1 \to \mathsf{ans},1}^{(t)} - \mathbf{Attn}_{\mathsf{ans},1 \to \mathsf{pred},2}^{(t)}\big)\Lambda_{5,j,r}^{(t)} \pm \widetilde{O}(\sigma_0)\Big)$$

$$\geq \big(1 - \mathbf{Attn}_{\mathsf{ans},1 \to \mathsf{ans},1}^{(t)} - \mathbf{Attn}_{\mathsf{ans},1 \to \mathsf{pred},2}^{(t)}\big)\Lambda_{5,j,r}^{(t)}.$$

- for $j = g_1(y)$, where $\exists y \neq y_0, y_1$, s.t., $g_2(y) = g_1(y)$, we have

$$\Big(\mathbf{Attn}_{\mathsf{ans},1 \to \mathsf{ans},0}^{(t)} \cdot V_{j,r}(y_0) + \mathbf{Attn}_{\mathsf{ans},1 \to \mathsf{pred},1}^{(t)} \cdot V_{j,r}(g_1)$$

$$- \big(1 - \mathbf{Attn}_{\mathsf{ans},1 \to \mathsf{ans},1}^{(t)} - \mathbf{Attn}_{\mathsf{ans},1 \to \mathsf{pred},2}^{(t)}\big)\Lambda_{5,j,r}^{(t)} \pm \widetilde{O}(\sigma_0)\Big)$$

$$\geq -\big(1 - \mathbf{Attn}_{\mathsf{ans},1 \to \mathsf{ans},1}^{(t)} - \mathbf{Attn}_{\mathsf{ans},1 \to \mathsf{pred},2}^{(t)}\big)\Lambda_{5,j,r}^{(t)}.$$

Putting them together, and upper bound $\sum_{j \neq j_2 \in \tau(\mathcal{Y})} \mathbf{logit}_{5,j}^{(t)}$ by $1 - \mathbf{logit}_{5,j_2}^{(t)}$, we have

$$\mathbb{E}\Big[\mathcal{N}_{s,2,ii}^{(t)} \mathbb{1}_{\mathcal{E}_1}\Big]$$

$$= \mathbb{E}\Bigg[\sum_{j \neq j_2 \in \tau(\mathcal{Y})} \mathbf{logit}_{5,j}^{(t)} \cdot \Big(\sum_{r \in \widehat{\mathfrak{A}}_j} \mathbf{sReLU}'(\Lambda_{5,j,r}) \cdot$$

$$\Big(\mathbf{Attn}_{\mathsf{ans},1 \to \mathsf{ans},0}^{(t)} \cdot V_{j,r}(y_0) + \mathbf{Attn}_{\mathsf{ans},1 \to \mathsf{pred},1}^{(t)} \cdot V_{j,r}(g_1)$$

$$- \big(1 - \mathbf{Attn}_{\mathsf{ans},1 \to \mathsf{ans},1}^{(t)} - \mathbf{Attn}_{\mathsf{ans},1 \to \mathsf{pred},2}^{(t)}\big)\Lambda_{5,j,r}^{(t)} \pm \widetilde{O}(\sigma_0)\Big) \pm \widetilde{O}(\delta^q)\Big) \mathbb{1}_{\tau(x_1)=s} \mathbb{1}_{\mathcal{E}_1}\Bigg]$$

$$\geq -\mathbb{E}\Bigg[\Big(1 - \mathbf{logit}_{5,j_2}^{(t)}\Big) \cdot \Big(1 - \mathbf{Attn}_{\mathrm{ans},1\to\mathrm{ans},1}^{(t)} - \mathbf{Attn}_{\mathrm{ans},1\to\mathrm{pred},2}^{(t)}\Big)$$

$$\cdot \max_{y\neq y_0, y_1}\Big\{\Lambda_{5,\tau(g_1(y_1)),r_{g_1\cdot y_1}}^{(t)}, \Lambda_{5,\tau(g_2(y_0)),r_{g_2\cdot y_0}}^{(t)}\Big\}\mathbb{1}_{\tau(x_1)=s}\mathbb{1}_{\mathcal{E}_1}\Bigg],$$

where the last inequality is due to the fact that $\Lambda_{5,\tau(g_2(y_0)),r_{g_2\cdot y_0}}^{(t)} = \Lambda_{5,\tau(g_2(y)),r_{g_2\cdot y}}^{(t)} \pm \widetilde{O}(\frac{B}{d\cdot n_y})$ for $y \neq y_0, y_1$ s.t., $g_1(y) = g_2(y)$. Notice that

$$\max_{y\neq y_0,y_1}\Big\{\Lambda_{5,\tau(g_1(y_1)),r_{g_1\cdot y_1}}^{(t)}, \Lambda_{5,\tau(g_2(y_0)),r_{g_2\cdot y_0}}^{(t)}\Big\} \leq$$

$$\max\Big\{\mathbf{Attn}_{\mathrm{ans},1\to\mathrm{pred},2}^{(t)} - \mathbf{Attn}_{\mathrm{ans},1\to\mathrm{ans},1}^{(t)}, \Theta(\frac{1}{n_y})\Big\}2B,$$

while

$$(110) \geq \mathbb{E}\Bigg[\Big(1 - \mathbf{logit}_{5,j_2}^{(t)}\Big) \cdot \mathbf{Attn}_{\mathrm{ans},1\to\mathrm{ans},0}^{(t)} \cdot 2B\mathbb{1}_{\tau(x_1)=s}\mathbb{1}_{\mathcal{E}_1}\Bigg]$$

$$= \frac{1}{2}\mathbb{E}\Bigg[\Big(1 - \mathbf{logit}_{5,j_2}^{(t)}\Big) \cdot \Big(1 - \mathbf{Attn}_{\mathrm{ans},1\to\mathrm{ans},1}^{(t)} - \mathbf{Attn}_{\mathrm{ans},1\to\mathrm{pred},2}^{(t)}\Big)2B\mathbb{1}_{\tau(x_1)=s}\mathbb{1}_{\mathcal{E}_1}\Bigg].$$

Since by Induction H.1, $\mathbf{Attn}_{\mathrm{ans},1\to\mathrm{pred},2}^{(t)} - \mathbf{Attn}_{\mathrm{ans},1\to\mathrm{ans},1}^{(t)} \leq c_1 \ll \frac{1}{2}$, thus we have

$$\mathbb{E}\Big[\mathcal{N}_{s,2,i}^{(t)}\mathbb{1}_{\mathcal{E}_1}\Big] + \mathbb{E}\Big[\mathcal{N}_{s,2,ii}^{(t)}\mathbb{1}_{\mathcal{E}_1}\Big] \geq \Omega\Big(\frac{B}{d}\Big) \cdot \mathbb{E}\Big[(1 - \mathbf{logit}_{5,j_2}^{(t)})\big|\tau(x_1) = s, \mathcal{E}_1\Big].$$

(b) For $\mathbf{Z}^{2,1} \in \mathcal{E}_2$, $\mathcal{N}_{s,2,i}^{(t)}$ can be bounded in the same way as (110), and we have

$$\mathbb{E}\Big[\mathcal{N}_{s,2,i}^{(t)}\mathbb{1}_{\mathcal{E}_2}\Big] \geq \Omega\Big(\frac{B}{d\cdot n_y}\Big) \cdot \mathbb{E}\Big[(1 - \mathbf{logit}_{5,j_2}^{(t)})\big|\tau(x_1) = s, \mathcal{E}_2\Big] \geq 0.$$

Moving to $\mathcal{N}_{s,2,ii}^{(t)}$,

- for $j = \tau(g_1(y_1))$, we have

$$= \mathbb{E}\Bigg[\mathbf{logit}_{5,j}^{(t)}\Big(\mathbf{sReLU}'(\Lambda_{5,j,r_{g_1\cdot y_1}})\cdot$$

$$\Big(\mathbf{Attn}_{\mathrm{ans},1\to\mathrm{ans},0}^{(t)} \cdot V_{j,r_{g_1\cdot y_1}}(y_0) + \mathbf{Attn}_{\mathrm{ans},1\to\mathrm{pred},1}^{(t)} \cdot V_{j,r_{g_1\cdot y_1}}(g_1) \qquad (111)$$

$$- \big(1 - \mathbf{Attn}_{\mathrm{ans},1\to\mathrm{ans},1}^{(t)} - \mathbf{Attn}_{\mathrm{ans},1\to\mathrm{pred},2}^{(t)}\big)\Lambda_{5,j,r}^{(t)} \pm \widetilde{O}(\sigma_0)\Big) \pm \widetilde{O}(\delta^q)\Big)\mathbb{1}_{\tau(x_1)=s}\mathbb{1}_{\mathcal{E}_2}\Bigg]$$

$$\geq -O\Big(\frac{B}{n_y^2\cdot d}\Big) \cdot \mathbb{E}\Big[(1 - \mathbf{logit}_{5,j_2}^{(t)})\big|\tau(x_1) = s, \mathcal{E}_2\Big].$$

where the last inequality is due to the cancellation of the term $\mathbf{Attn}_{\mathrm{ans},1\to\mathrm{ans},0}^{(t)} \cdot V_{j,r_{g_1\cdot y_1}}(y_0) + \mathbf{Attn}_{\mathrm{ans},1\to\mathrm{pred},1}^{(t)} \cdot V_{j,r_{g_1\cdot y_1}}(g_1)$ and the fact that

$$\Lambda_{5,\tau(g_1(y_1)),r_{g_1\cdot y_1}}^{(t)} = \Big(\mathbf{Attn}_{\mathrm{ans},1\to\mathrm{ans},1}^{(t)} - \mathbf{Attn}_{\mathrm{ans},1\to\mathrm{pred},2}^{(t)}\Big) \cdot \frac{2B}{n_y} + O(\delta) \leq O\big(\frac{B}{n_y}\big).$$

- for $j_2' = \tau(g_2(y_0)) = \tau(g_1(y_0))$, clearly, we have

$$\mathbb{E}\Bigg[\mathbf{logit}_{5,j_2'}^{(t)} \cdot \Big(\mathbf{sReLU}'(\Lambda_{5,j_2',r_{g_1\cdot y_0}}^{(t)})\cdot$$

$$\Big(\mathbf{Attn}_{\mathrm{ans},1\to\mathrm{ans},0}^{(t)} \cdot V_{j_2',r_{g_1\cdot y_0}}(y_0) + \mathbf{Attn}_{\mathrm{ans},1\to\mathrm{pred},1}^{(t)} \cdot V_{j_2',r_{g_1\cdot y_0}}(g_1)$$

$$- \big(1 - \mathbf{Attn}_{\mathrm{ans},1\to\mathrm{ans},1}^{(t)} - \mathbf{Attn}_{\mathrm{ans},1\to\mathrm{pred},2}^{(t)}\big)\Lambda_{5,j_2',r_{g_1\cdot y_0}}^{(t)} \pm \widetilde{O}(\sigma_0)\Big) \pm \widetilde{O}(\delta^q)\Big)$$

$$\mathbb{1}_{\tau(x_1)=s}\mathbb{1}_{\mathcal{E}_2}\Big] \geq 0,$$

where the last inequality is due to the fact that

$$\mathbf{Attn}^{(t)}_{\mathsf{ans},1\to\mathsf{pred},1} \cdot V_{j_2',r_{g_1\cdot y_0}}(g_1) - \big(1 - \mathbf{Attn}^{(t)}_{\mathsf{ans},1\to\mathsf{ans},1}$$
$$- \mathbf{Attn}^{(t)}_{\mathsf{ans},1\to\mathsf{pred},2}\big)\Lambda^{(t)}_{5,j_2',r_{g_1\cdot y_0}}$$

$$\geq \mathbf{Attn}^{(t)}_{\mathsf{ans},1\to\mathsf{pred},1}$$
$$\cdot \Big(1 - 2\big(\mathbf{Attn}^{(t)}_{\mathsf{ans},1\to\mathsf{pred},2} + \mathbf{Attn}^{(t)}_{\mathsf{ans},1\to\mathsf{pred},1} - \mathbf{Attn}^{(t)}_{\mathsf{ans},1\to\mathsf{ans},1}\big)\Big)2B$$

$$\geq \mathbf{Attn}^{(t)}_{\mathsf{ans},1\to\mathsf{pred},1}\Big(1 - 2\big(c_1 + 0.25\big)\Big)2B \geq 0.$$

- for $j = \tau(g_2(y))$ if there exists $y \neq y_0, y_1$ s.t., $g_1(y) = g_2(y)$
  - when $\mathbb{E}\big[\mathbf{Attn}^{(t)}_{\mathsf{ans},1\to\mathsf{ans},1} - \mathbf{Attn}^{(t)}_{\mathsf{ans},1\to\mathsf{pred},2} \mid \tau(x_1) = s\big] \leq 0.01$, by Induction H.1 and Lemma H.14, we have $\mathbf{logit}^{(t)}_{5,\tau(g_2(y))} \ll O(\frac{1}{d}) \cdot \mathbf{logit}^{(t)}_{5,j_2'}$. Then

$$\bigg|\mathbb{E}\bigg[\mathbf{logit}^{(t)}_{5,j} \cdot \Big(\mathbf{sReLU}'(\Lambda^{(t)}_{5,j,r_{g_1\cdot y}})\cdot$$
$$\Big(\mathbf{Attn}^{(t)}_{\mathsf{ans},1\to\mathsf{ans},0} \cdot V_{j,r_{g_1\cdot y}}(y_0) + \mathbf{Attn}^{(t)}_{\mathsf{ans},1\to\mathsf{pred},1} \cdot V_{j,r_{g_1\cdot y}}(g_1)$$
$$- \big(1 - \mathbf{Attn}^{(t)}_{\mathsf{ans},1\to\mathsf{ans},1} - \mathbf{Attn}^{(t)}_{\mathsf{ans},1\to\mathsf{pred},2}\big)\Lambda^{(t)}_{5,j,r_{g_1\cdot y}} \pm \widetilde{O}(\sigma_0)\Big) \pm \widetilde{O}(\delta^q)\Big)\mathbb{1}_{\tau(x_1)=s}\mathbb{1}_{\mathcal{E}_2}\bigg]\bigg|$$

$$\leq O\Big(\frac{1}{d}\Big) \cdot (111).$$

  - when $\mathbb{E}\big[\mathbf{Attn}^{(t)}_{\mathsf{ans},1\to\mathsf{ans},1} - \mathbf{Attn}^{(t)}_{\mathsf{ans},1\to\mathsf{pred},2} \mid \tau(x_1) = s\big] \geq 0.01$, by Lemma H.14, we have $\Lambda^{(t)}_{5,j,r_{g_1\cdot y}}$ cannot be activated.

Putting the above discussion together, we have

$$\mathbb{E}\Big[\mathcal{N}^{(t)}_{s,2,ii}\mathbb{1}_{\mathcal{E}_1}\Big] + \mathbb{E}\Big[\mathcal{N}^{(t)}_{s,2,ii}\mathbb{1}_{\mathcal{E}_2}\Big] \geq \Omega\Big(\frac{B}{d \cdot n_y}\Big) \cdot \mathbb{E}\Big[(1 - \mathbf{logit}^{(t)}_{5,j_2})|\tau(x_1) = s, \mathcal{E}_2\Big].$$

(c) For $\mathbf{Z}^{2,1} \in \mathcal{E}_3$, by Induction H.1 and Lemma H.15, we can first have the following logit bound:

$$\mathbf{logit}^{(t)}_{5,j} \leq \frac{1}{\mathrm{poly}d} \cdot \Big(1 - \mathbf{logit}^{(t)}_{5,j_2}\Big) \text{ for } j \neq j_2, \tau\big(g_1(y_1)\big).$$

Therefore, we have

$$\mathbb{E}\bigg[\mathcal{N}^{(t)}_{s,2,i}\mathbb{1}_{\mathcal{E}_3}\bigg]$$

$$= \mathbb{E}\bigg[(1 - \mathbf{logit}^{(t)}_{5,j_2}) \cdot \Big(- \mathbf{Attn}^{(t)}_{\mathsf{ans},1\to\mathsf{ans},0} \cdot V_{j_2,r_{g_2\cdot y_1}}(y_1) - \mathbf{Attn}^{(t)}_{\mathsf{ans},1\to\mathsf{pred},1} \cdot V_{j_2,r_{g_2\cdot y_1}}(g_1)$$
$$+ \big(1 - \mathbf{Attn}^{(t)}_{\mathsf{ans},1\to\mathsf{ans},1} - \mathbf{Attn}^{(t)}_{\mathsf{ans},1\to\mathsf{pred},2}\big)\Lambda^{(t)}_{5,j_2,r_{g_2\cdot y_1}} \pm \widetilde{O}(\sigma_0) \pm \widetilde{O}(\delta^q)\Big)\mathbb{1}_{\tau(x_1)=s}\mathbb{1}_{\mathcal{E}_3}\bigg]$$

$$\geq \Omega\Big(\frac{B}{n_y \cdot d}\Big) \cdot \mathbb{E}\bigg[(1 - \mathbf{logit}^{(t)}_{5,j_2})|\tau(x_1) = s, \mathcal{E}_3\bigg].$$

Moving to $\mathcal{N}^{(t)}_{s,2,ii}$, we have

$$\mathbb{E}\bigg[\mathcal{N}^{(t)}_{s,2,ii}\mathbb{1}_{\mathcal{E}_3}\bigg]$$

$$\geq \mathbb{E}\Bigg[\mathbf{logit}^{(t)}_{5,\tau(g_1(y_1))} \cdot$$

$$\Big(\mathbf{Attn}^{(t)}_{\mathsf{ans},1\to\mathsf{ans},0} \cdot V_{\tau(g_1(y_1)),r_{g_1\cdot y_1}}(y_1) + \mathbf{Attn}^{(t)}_{\mathsf{ans},1\to\mathsf{pred},1} \cdot V_{\tau(g_1(y_1)),r}(g_1)$$

$$- \big(1 - \mathbf{Attn}^{(t)}_{\mathsf{ans},1\to\mathsf{ans},1} - \mathbf{Attn}^{(t)}_{\mathsf{ans},1\to\mathsf{pred},2}\big)\Lambda^{(t)}_{5,\tau(g_1(y_1)),r_{g_1\cdot y_1}} \pm \widetilde{O}(\sigma_0) \pm \widetilde{O}(\delta^q)\Big)$$

$$\mathbb{1}_{\tau(x_1)=s}\mathbb{1}_{\mathcal{E}_3}\Bigg]$$

$$- O\Big(\frac{n_y \cdot B}{n_y \cdot d \cdot \mathsf{poly}d}\Big) \cdot \mathbb{E}\Big[(1 - \mathbf{logit}^{(t)}_{5,j_2})\big|\tau(x_1) = s, \mathcal{E}_3\Big].$$

Thus,

$$\mathbb{E}\Big[\mathcal{N}^{(t)}_{s,2,ii}\mathbb{1}_{\mathcal{E}_3}\Big] + \mathbb{E}\Big[\mathcal{N}^{(t)}_{s,2,ii}\mathbb{1}_{\mathcal{E}_3}\Big] \geq \Omega\Big(\frac{B}{n_y \cdot d}\Big) \cdot \mathbb{E}\Big[(1 - \mathbf{logit}^{(t)}_{5,j_2})\big|\tau(x_1) = s, \mathcal{E}_3\Big].$$

(d) For $\mathbf{Z}^{2,1} \in \mathcal{E}_4$, we can bound the gradient in a manner similar to regime (i), and obtain

$$\mathbb{E}\Big[\mathcal{N}^{(t)}_{s,2,ii}\mathbb{1}_{\mathcal{E}_4}\Big] + \mathbb{E}\Big[\mathcal{N}^{(t)}_{s,2,ii}\mathbb{1}_{\mathcal{E}_4}\Big] \geq 0.$$

(e) For $\mathbf{Z}^{2,1} \in \mathcal{E}_5 \cup \mathcal{E}_6$, by Induction H.1, Lemma H.17 and Lemma H.18, we can derive the following logit condition: $1 - \mathbf{logit}^{(t)}_{5,j_2}, \mathbf{logit}^{(t)}_{5,j} \leq \frac{1}{\mathsf{poly}d} \cdot \mathbb{E}\Big[(1 - \mathbf{logit}^{(t)}_{5,j_2})\big|\tau(x_1) = s, \mathcal{E}_1\Big]$ for $j \neq j_2$. Hence, we can simply bound $\mathcal{N}^{(t)}_{s,2,i}$ and $\mathcal{N}^{(t)}_{s,2,ii}$ as follows:

$$\Big|\mathbb{E}\Big[\mathcal{N}^{(t)}_{s,2,i}\mathbb{1}_{\mathcal{E}_m}\Big]\Big|, \Big|\mathbb{E}\Big[\mathcal{N}^{(t)}_{s,2,ii}\mathbb{1}_{\mathcal{E}_m}\Big]\Big|$$

$$\leq O\Big(\frac{B}{dn_y} \cdot \frac{1}{\mathsf{poly}d}\Big) \cdot \mathbb{E}\Big[(1 - \mathbf{logit}^{(t)}_{5,j_2})\big|\tau(x_1) = s, \mathcal{E}_1\Big], \text{ for } m \in \{5, 6\}.$$

Putting everything together, we have for the regmie (ii),

$$\Big[-\nabla_{\mathbf{Q}^{(t)}_{4,3}}\mathsf{Loss}^{2,2}_5\Big]_{s,s} + \Big[-\nabla_{\mathbf{Q}^{(t)}_{4,4}}\mathsf{Loss}^{2,2}_5\Big]_{s,s} = \sum_{\kappa\in\{i,ii,iii\}}\sum_{i\in[6]}\mathbb{E}\Big[\mathcal{N}^{(t)}_{s,2,\kappa}\mathbb{1}_{\mathcal{E}_i}\Big]$$

$$\geq \Omega\Big(\frac{B}{d}\Big) \cdot \mathbb{E}\Big[(1 - \mathbf{logit}^{(t)}_{5,j_2})\big|\tau(x_1) = s, \mathcal{E}_1\Big].$$

$\square$

**Lemma H.20.** *If Induction H.1 holds for all iterations $< t$, given $s \in \tau(\mathcal{X})$, for $[\mathbf{Q}_{4,p}]_{s,s'}$, $p \in \{3, 4\}$, $s' \neq s \in \tau(\mathcal{X})$, we have*

$$\Big|\Big[-\nabla_{\mathbf{Q}^{(t)}_{4,p}}\mathsf{Loss}^{2,2}_5\Big]_{s,s'}\Big| \leq O\Big(\frac{1}{d}\Big)\Big|\Big[-\nabla_{\mathbf{Q}^{(t)}_{4,p}}\mathsf{Loss}^{2,2}_5\Big]_{s,s}\Big|.$$

### H.3.3 Bounded Decrease of Attention to Related Context Clause

**Lemma H.21.** *If Induction H.1 holds for all iterations $< t$, then for any sample $\mathbf{Z}^{2,1}$, we have*

$$\mathbf{Attn}^{(t)}_{\mathsf{ans},1\to\mathsf{pred},1} - \mathbf{Attn}^{(t)}_{\mathsf{ans},1\to\mathsf{pred},2} \geq -O\Big(\frac{\log d \log d}{\log d}\Big).$$

*Proof.* Denote the first time that $\mathbb{E}\Big[\mathbf{Attn}^{(t)}_{\mathsf{ans},1\to\mathsf{pred},1} - \mathbf{Attn}^{(t)}_{\mathsf{ans},1\to\mathsf{pred},2}\big|\tau(x_1) = s\Big] \leq -\Omega\big(\frac{\log d \log d}{\log d}\big)$ as $\widetilde{T}$. Notice that $\Big|\big[\mathbf{Q}^{(t)}_{4,p}\big]_{s,s'}\Big| \leq O\Big(\frac{\big[\mathbf{Q}^{(t)}_{4,p}\big]_{s,s}}{d}\Big)$ for $p \in \{3, 4\}$, thus for any

sample $\mathbf{Z}^{2,1}$ satisfying $\tau(x_1) = s$, at time $\widetilde{T}$, we have

$$\mathbf{Attn}^{(\widetilde{T})}_{\text{ans},1\rightarrow\text{pred},1} - \mathbf{Attn}^{(\widetilde{T})}_{\text{ans},1\rightarrow\text{pred},2} \leq -\Omega\Big(\frac{\log d \log d}{\log d}\Big).$$

Based on the the gradient compositions from Lemma H.11, we have $\Big[ -\nabla_{\mathbf{Q}_{4,3}} \text{Loss}_5^{2,2} \Big]_{s,s} = \mathbb{E}\Big[\mathcal{N}_{s,3,2,i} + \mathcal{N}_{s,3,2,ii} + \mathcal{N}_{s,3,2,iii}\Big]$, and we will discuss $\mathcal{N}_{s,3,2,\kappa}$ for $\kappa \in \{i, ii, iii\}$ on different samples $\mathbf{Z}^{2,1}$. Following the similar argument as Lemma H.19, we can first directly bound the term $\mathcal{N}^{(\widetilde{T})}_{s,3,2,iii}$ as follows:

$$\mathbb{E}\Big[\mathcal{N}^{(\widetilde{T})}_{s,3,2,iii}\Big] \leq \widetilde{O}(\sigma_0^q) = \frac{1}{\text{poly}d}.$$

- for $\mathbf{Z}^{2,1} \in \mathcal{E}_1$, at time $\widetilde{T}$, since the neurons for predicting $j_2$ cannot be activated, and $\mathbf{logit}^{(\widetilde{T})}_{5,j} \leq O(\frac{1}{d})$ for $j \neq j_2$, thus we can naively bound the gradient on the event $\mathcal{E}_1$ as follows:

$$\Big| \sum_{\kappa \in \{i, ii\}} \mathbb{E}\big[\mathcal{N}^{(\widetilde{T})}_{s,3,2,\kappa} \mathbb{1}_{\mathcal{E}_1}\big] \Big| \leq O\Big(\frac{n_y B}{d^2}\Big) + \widetilde{O}(\delta^q).$$

- for $\mathbf{Z}^{2,1} \in \mathcal{E}_2$, similarly, $\Lambda^{(t)}_{5,j_2,r}$ is not activated, and thus we can focus on the term $\mathcal{N}_{s,3,2,ii}$, and specifically, the prediction of $j_2' = \tau(g_2(y_0))$. By (98) and Lemma H.14, we have

$$\mathbb{E}\Big[\mathcal{N}^{(\widetilde{T})}_{s,3,2,ii} \mathbb{1}_{\mathcal{E}_2}\Big] \geq -\mathbb{E}\Big[\mathbf{Attn}^{(\widetilde{T})}_{\text{ans},1\rightarrow\text{pred},2}$$
$$\cdot \Big(1 + \mathbf{Attn}^{(\widetilde{T})}_{\text{ans},1\rightarrow\text{ans},1} - \mathbf{Attn}^{(\widetilde{T})}_{\text{ans},1\rightarrow\text{pred},2} - \mathbf{Attn}^{(\widetilde{T})}_{\text{ans},1\rightarrow\text{pred},1}\Big) \mid \tau(x_1) = s\Big]$$
$$\cdot \mathbb{E}\Big[\mathbf{logit}^{(\widetilde{T})}_{5,j_2'} \mid \tau(x_1) = s, \mathcal{E}_2\Big]\frac{2B}{n_y \cdot d} \pm \widetilde{O}(\sigma_0^q). \tag{112}$$

- for $\mathbf{Z}^{2,1} \in \mathcal{E}_3$, by Lemma H.15, we can mainly focus on the term $\mathcal{N}^{(\widetilde{T})}_{s,3,2,i}$, and the prediction of $\tau(g_1(y_1))$ in $\mathcal{N}^{(\widetilde{T})}_{s,3,2,ii}$ since $\mathbf{logit}^{(\widetilde{T})}_{5,\tau(g_1(y_1))} = 1 - O\big(\frac{1}{\log d}\big)$. Hence, we have

$$\mathbb{E}\Big[\big(\mathcal{N}^{(\widetilde{T})}_{s,3,2,ii} + \mathcal{N}^{(\widetilde{T})}_{s,3,2,ii}\big)\mathbb{1}_{\mathcal{E}_3}\Big] \geq \mathbb{E}\Big[\mathbf{Attn}^{(\widetilde{T})}_{\text{ans},1\rightarrow\text{pred},2} \tag{113}$$
$$\cdot \Big(1 + \mathbf{Attn}^{(\widetilde{T})}_{\text{ans},1\rightarrow\text{pred},1} - \mathbf{Attn}^{(\widetilde{T})}_{\text{ans},1\rightarrow\text{pred},2}\Big) \mid \tau(x_1) = s\Big] \cdot \Big(1 - O\big(\frac{1}{\log d}\big)\Big)\frac{2B}{n_y \cdot d} \pm \widetilde{O}(\sigma_0^q).$$

- for $\mathbf{Z}^{2,1} \in \mathcal{E}_4$, the negative gradient can be bounded in the same way as (112), however, the probability of $\mathcal{E}_4$ is order-wise smaller than $\mathcal{E}_2$ and $\mathcal{E}_3$, which can be neglected. Moreover, for for $\mathbf{Z}^{2,1} \in \mathcal{E}_5 \cup \mathcal{E}_6$, the overall gradient is also negeligble since $\mathbf{logit}^{(t)}_{5,j_2}$ is very close to 1.

Putting it all together, we have

$$\Big[ -\nabla_{\mathbf{Q}^{(\widetilde{T})}_{4,3}} \text{Loss}_5^{2,2} \Big]_{s,s} \geq -\mathbb{E}\Big[\mathbf{Attn}^{(\widetilde{T})}_{\text{ans},1\rightarrow\text{pred},2}$$
$$\cdot \Big(1 + \mathbf{Attn}^{(\widetilde{T})}_{\text{ans},1\rightarrow\text{ans},1} - \mathbf{Attn}^{(\widetilde{T})}_{\text{ans},1\rightarrow\text{pred},2} - \mathbf{Attn}^{(\widetilde{T})}_{\text{ans},1\rightarrow\text{pred},1}\Big) \mid \tau(x_1) = s\Big]$$
$$\cdot \mathbb{E}\Big[\mathbf{logit}^{(\widetilde{T})}_{5,j_2'} \mid \tau(x_1) = s, \mathcal{E}_2\Big]\frac{B}{n_y \cdot d}$$
$$+ \mathbb{E}\Big[\mathbf{Attn}^{(\widetilde{T})}_{\text{ans},1\rightarrow\text{pred},2}$$

$$\cdot \left(1 + \mathbf{Attn}^{(\widetilde{T})}_{\text{ans},1\to\text{pred},1} - \mathbf{Attn}^{(\widetilde{T})}_{\text{ans},1\to\text{pred},2}\right) \mid \tau(x_1) = s\right] \cdot \left(1 - O(\frac{1}{\log d})\right)\frac{2B}{n_y \cdot d} \pm \widetilde{O}(\sigma_0^q).$$

Notice that

$$\left(1 + \mathbf{Attn}^{(\widetilde{T})}_{\text{ans},1\to\text{pred},1} - \mathbf{Attn}^{(\widetilde{T})}_{\text{ans},1\to\text{pred},2}\right)$$

$$- \left(1 + \mathbf{Attn}^{(\widetilde{T})}_{\text{ans},1\to\text{ans},1} - \mathbf{Attn}^{(\widetilde{T})}_{\text{ans},1\to\text{pred},2} - \mathbf{Attn}^{(\widetilde{T})}_{\text{ans},1\to\text{pred},1}\right)$$

$$= 2\mathbf{Attn}^{(\widetilde{T})}_{\text{ans},1\to\text{pred},1} - \mathbf{Attn}^{(\widetilde{T})}_{\text{ans},1\to\text{ans},1}.$$

If $2\mathbf{Attn}^{(\widetilde{T})}_{\text{ans},1\to\text{pred},1} - \mathbf{Attn}^{(\widetilde{T})}_{\text{ans},1\to\text{ans},1} < c$ for some small constant $c > 0$, s.t., $\frac{cB}{\log d} < 1$, then $\Lambda^{(\widetilde{T})}_{5,j_2',r_{g_2 \cdot y_0}} \leq cB$. Hence $\mathbf{logit}^{(\widetilde{T})}_{5,j_2'} \leq \frac{1}{d^{1-\frac{cB}{\log d}}} = o(1)$, and (112) is dominated by the positive term (113). Else, clearly, (112) can be cancelled out by the positive term (113). Therefore,

$$\left[-\nabla_{\mathbf{Q}^{(\widetilde{T})}_{4,3}}\mathsf{Loss}_5^{2,2}\right]_{s,s} \geq \Omega\left(\frac{B}{n_y \cdot d}\right).$$

As a consequence, together with the growth of $\mathbf{Q}^{(\widetilde{T})}_{4,3} + \mathbf{Q}^{(\widetilde{T})}_{4,4}$, and nearly no change of $[\mathbf{Q}^{(t)}_{4,p}]_{s,s'}$ in Lemma H.20, we have $\mathbf{Attn}^{(t)}_{\text{ans},1\to\text{pred},1} - \mathbf{Attn}^{(t)}_{\text{ans},1\to\text{pred},2}$ must start to decrease and cannot be larger than $O\left(\frac{\log\log d}{\log d}\right)$. $\qquad\square$

### H.3.4 Attention gap is small

**Lemma H.22.** *If Induction H.1 holds for all iterations $< t$, then for any sample $\mathbf{Z}^{2,1}$, we have*

$$\mathbf{Attn}^{(t)}_{\text{ans},1\to\text{pred},2} - \mathbf{Attn}^{(t)}_{\text{ans},1\to\text{ans},1} \leq c_1,$$

*where $c_1 > 0$ is a small constant.*

*Proof.* Let $\widetilde{T}$ denote the first time $\mathbb{E}\left[\mathbf{Attn}^{(t)}_{\text{ans},1\to\text{pred},2} - \mathbf{Attn}^{(t)}_{\text{ans},1\to\text{ans},1}\big|\tau(x_1) = s\right] \geq \frac{1.0005\log d}{2B}$.

Notice that $\left|\left[\mathbf{Q}^{(t)}_{4,p}\right]_{s,s'}\right| \leq O\left(\frac{[\mathbf{Q}^{(t)}_{4,p}]_{s,s}}{d}\right)$ for $p \in \{3,4\}$, thus for any sample $\mathbf{Z}^{2,1}$ satisfying $\tau(x_1) = s$, at time $\widetilde{T}$, we have

$$\mathbf{Attn}^{(\widetilde{T})}_{\text{ans},1\to\text{pred},2} - \mathbf{Attn}^{(\widetilde{T})}_{\text{ans},1\to\text{ans},1} \geq \frac{1.0005\log d}{2B}.$$

Based on Lemma H.11, we will discuss the gradients of $\mathbf{Q}_{4,3}$ and $\mathbf{Q}_{4,4}$ on different samples $\mathbf{Z}^{2,1}$. Since $\Lambda^{(\widetilde{T})}_{5,j_2,r_{g_2 \cdot y_1}}, \Lambda^{(\widetilde{T})}_{5,j_2',r_{g_2 \cdot y_0}}$ is guaranteed to be activated to the linear regime for $\mathbf{Z}^{2,1} \in \mathcal{E}_1$, we only need to focus on the main event $\mathcal{E}_1$.

From Lemma H.13, at time $\widetilde{T}$, we have

$$\mathbf{logit}^{(\widetilde{T})}_{5,j_2'} = \frac{e^{\frac{1.0005\log d}{2B}\cdot 2B}}{e^{\frac{1.0005\log d}{2B}\cdot 2B} + O(d)}\left(1 - \mathbf{logit}^{(\widetilde{T})}_{5,j_2}\right) = \left(1 - O(1/d^{0.0005})\right)\left(1 - \mathbf{logit}^{(\widetilde{T})}_{5,j_2}\right).$$

Then by Lemma H.11, we can obtain the gradient on the event $\mathcal{E}_1$ as follows:

$$\mathbb{E}\left[\left(\mathcal{N}^{(\widetilde{T})}_{s,3,2,i} + \mathcal{N}^{(\widetilde{T})}_{s,3,2,ii} + \mathcal{N}^{(\widetilde{T})}_{s,3,2,iii}\right)\mathbb{1}_{\mathcal{E}_1}\right]$$

$$= \mathbb{E}\left[\mathbf{Attn}^{(\widetilde{T})}_{\text{ans},1\to\text{pred},2} \cdot \left(\left(1 - \mathbf{logit}^{(\widetilde{T})}_{5,j_2}\right)\cdot\left(V_{j_2,r_{g_2 \cdot y_1}}(g_2) - \Lambda^{(\widetilde{T})}_{5,j_2,r_{g_2 \cdot y_1}} \pm \widetilde{O}(\sigma_0)\right)\right.\right.$$

$$\left.\left. - \left(1 - O\left(\frac{1}{d^{0.0005}}\right)\right)(1 - \mathbf{logit}^{(t)}_{5,j_2})\cdot\left(V_{j_2',r_{g_2 \cdot y_0}}(g_2) - \Lambda^{(\widetilde{T})}_{5,j_2,r_{g_2 \cdot y_0}} \pm \widetilde{O}(\sigma_0)\right)\right.\right.$$

$$\pm \widetilde{O}(\sigma_0^q)\Big)\mathbb{1}_{\tau(x_1)=s}\mathbb{1}_{\mathcal{E}_1}\Bigg]$$

$$= \mathbb{E}\Bigg[\mathbf{Attn}^{(\widetilde{T})}_{\mathrm{ans},1\to\mathrm{pred},2}\cdot\Big(\Big(1-O\Big(\frac{1}{d^{0.0005}}\Big)\Big)(1-\mathbf{logit}^{(\widetilde{T})}_{5,j_2})\cdot\Big(\Lambda^{(\widetilde{T})}_{5,j_2',r_{g_2\cdot y_0}}-\Lambda^{(\widetilde{T})}_{5,j_2,r_{g_2\cdot y_1}}\Big)$$

$$+\,O\Big(\frac{1}{d^{0.0005}}\Big)(1-\mathbf{logit}^{(\widetilde{T})}_{5,j_2})\cdot\Big(V_{5,j_2,r_{g_2\cdot y_1},2}(g_2)-\Lambda^{(\widetilde{T})}_{5,j_2,r_{g_2\cdot y_1}}\Big)\Big)\mathbb{1}_{\tau(x_1)=s}\mathbb{1}_{\mathcal{E}_1}\Bigg],$$

on the other hand, we have

$$\mathbb{E}\Bigg[\Big(\mathcal{N}^{(\widetilde{T})}_{s,4,2,i}+\mathcal{N}^{(\widetilde{T})}_{s,4,2,ii}+\mathcal{N}^{(\widetilde{T})}_{s,4,2,iii}\Big)\mathbb{1}_{\mathcal{E}_1}\Bigg]$$

$$= \mathbb{E}\Bigg[\mathbf{Attn}^{(\widetilde{T})}_{\mathrm{ans},1\to\mathrm{pred},2}\cdot\Big(\Big(1-\mathbf{logit}^{(\widetilde{T})}_{5,j_2}\Big)\cdot\Big(V_{j_2,r_{g_2\cdot y_1}}(y_1)-\Lambda^{(\widetilde{T})}_{5,j_2,r_{g_2\cdot y_1}}\pm\widetilde{O}(\sigma_0)\Big)$$

$$-\Big(1-O\Big(\frac{1}{d^{0.0005}}\Big)\Big)(1-\mathbf{logit}^{(t)}_{5,j_2})\cdot\Big(V_{j_2',r_{g_2\cdot y_0}}(y_1)-\Lambda^{(\widetilde{T})}_{5,j_2,r_{g_2\cdot y_0}}\pm\widetilde{O}(\sigma_0)\Big)$$

$$\pm\,\widetilde{O}(\sigma_0^q)\Big)\mathbb{1}_{\tau(x_1)=s}\mathbb{1}_{\mathcal{E}_1}\Bigg]$$

$$= \mathbb{E}\Bigg[\mathbf{Attn}^{(\widetilde{T})}_{\mathrm{ans},1\to\mathrm{pred},2}\cdot\Big(\Big(1-O\Big(\frac{1}{d^{0.0005}}\Big)\Big)(1-\mathbf{logit}^{(\widetilde{T})}_{5,j_2})$$

$$\cdot\Big(-V_{j_2',r_{g_2\cdot y_0}}(y_1)+\Lambda^{(\widetilde{T})}_{5,j_2',r_{g_2\cdot y_0}}-\Lambda^{(\widetilde{T})}_{5,j_2,r_{g_2\cdot y_1}}\Big)$$

$$-\,O\Big(\frac{1}{d^{0.0005}}\Big)(1-\mathbf{logit}^{(\widetilde{T})}_{5,j_2})\cdot\Lambda^{(\widetilde{T})}_{5,j_2,r_{g_2\cdot y_1}}\Big)\mathbb{1}_{\tau(x_1)=s}\mathbb{1}_{\mathcal{E}_1}\Bigg].$$

Since at time $\widetilde{T}$, we have $\mathbf{Attn}^{(\widetilde{T})}_{\mathrm{ans},1\to\mathrm{pred},1}-\mathbf{Attn}^{(\widetilde{T})}_{\mathrm{ans},1\to\mathrm{ans},1}\leq\mathbf{Attn}^{(\widetilde{T})}_{\mathrm{ans},1\to\mathrm{pred},2}-$ $\mathbf{Attn}^{(\widetilde{T})}_{\mathrm{ans},1\to\mathrm{ans},1}\leq\frac{1.005\log d}{2B}$, we have $\Lambda^{(\widetilde{T})}_{5,j_2',r_{g_2\cdot y_0}}-\Lambda^{(\widetilde{T})}_{5,j_2,r_{g_2\cdot y_1}}\ll -V_{j_2',r_{g_2\cdot y_0}}(y_1)$. Therefore,

$$\Big[-\nabla_{\mathbf{Q}^{(\widetilde{T})}_{4,4}}\mathsf{Loss}^{2,2}_5\Big]_{s,s}+\Big[\nabla_{\mathbf{Q}^{(\widetilde{T})}_{4,3}}\mathsf{Loss}^{2,2}_5\Big]_{s,s}\geq\Omega\Big(\frac{B}{d}\Big)\cdot\mathbb{E}\Big[1-\mathbf{logit}^{(\widetilde{T})}_{5,j_2}\big|\tau(x_1)=s\Big],$$

which implies that $[\mathbf{Q}_{4,4}]_{s,s}$ will grow faster than $[\mathbf{Q}_{4,3}]_{s,s}$, and thus the attention gap cannot be further increased. $\qquad\square$

### H.3.5   At the End of Stage 1.2.1

**Lemma H.23.** *For all iterations* $t\leq T_{1,2,1,s}=O\Big(\frac{n_y d}{\eta\log d}\Big)+O\Big(d^{\frac{(0.1+c_1/2)B}{\log d}}\Big)$, *we have Induction H.1 holds, and at time* $T_{1,2,1,s}$, *we have*

(a) $[\mathbf{Q}^{(T_{1,2,1,s})}_{4,3}]_{s,s},[\mathbf{Q}^{(T_{1,2,1,s})}_{4,4}]_{s,s}\geq\Omega(1)$;

(b) *other* $|[\mathbf{Q}^{(T_{2,1,s})}_{4,p}]_{s,s'}|$ *for* $p\in\{3,4\}$, $s'\in\tau(\mathcal{X})\neq s$ *are at most* $\widetilde{O}(\frac{1}{d})$.

*Proof.* The existence of $T_{1,2,1,s}$ can be directly obtained by using Lemmas H.19 to H.22. Furthermore, $[\mathbf{Q}^{(T_{1,2,1,s})}_{4,4}]_{s,s}\geq\Omega(1)$ can be guaranteed since Lemma H.22 implies that $\mathbf{Attn}^{(T_{1,2,1,s})}_{\mathrm{ans},1\to\mathrm{ans},1}\geq 0.3-\frac{c_1}{2}\pm\widetilde{O}(1/d)>0.2$, which means that $[\mathbf{Q}^{(T_{1,2,1,s})}_{4,4}]_{s,s}$ should at least grow to a constant level compared to $|[\mathbf{Q}^{(T_{1,2,1,s})}_{4,p}]_{s,s'}|=\widetilde{O}(1/d)$ with $p\in\{3,4\}$.

We will handle $[\mathbf{Q}^{(T_{1,2,1,s})}_{4,3}]_{s,s}$ by means of a proof by contradiction. Suppose that $\mathbf{Attn}^{(T_{1,2,1,s})}_{\mathrm{ans},1\to\mathrm{ans},1}\geq 0.4-\widetilde{c}$, where $\widetilde{c}=\frac{\log d}{8B}$ is a sufficiently small constant. Then denote $\widetilde{T}$ the first time that

$$\mathbb{E}\left[\mathbf{Attn}^{(t)}_{\mathsf{ans},1\to\mathsf{ans},1}\big|\tau(x_1)=s\right]\geq 0.4-2\widetilde{c}. \text{ Notice that } \left|\left[\mathbf{Q}^{(t)}_{4,p}\right]_{s,s'}\right|\leq O\left(\frac{\left[\mathbf{Q}^{(t)}_{4,p}\right]_{s,s}}{d}\right) \text{ for}$$

$p\in\{3,4\}$, thus for any sample $\mathbf{Z}^{2,1}$ satisfying $\tau(x_1)=s$, at time $\widetilde{T}$, we have

$$\mathbf{Attn}^{(\widetilde{T})}_{\mathsf{ans},1\to\mathsf{ans},1}\geq 0.4-2\widetilde{c}\pm\widetilde{O}(1/d).$$

- If $\mathbf{Attn}^{(\widetilde{T})}_{\mathsf{ans},1\to\mathsf{pred},2}-\mathbf{Attn}^{(\widetilde{T})}_{\mathsf{ans},1\to\mathsf{ans},0}\geq 2\varrho$, then

  - for $\mathbf{Z}^{2,1}\in\mathcal{E}_1$, $\Lambda^{(\widetilde{T})}_{5,j_2,r_{g_2\cdot y_1}}$ has already been activated to the linear regime. Furthermore, $\mathbf{Attn}^{(\widetilde{T})}_{\mathsf{ans},1\to\mathsf{pred},2}-\mathbf{Attn}^{(\widetilde{T})}_{\mathsf{ans},1\to\mathsf{ans},0}\leq 0.2+2\widetilde{c}-0.2\leq 2\widetilde{c}$, which implies $1-\mathbf{logit}^{(1)}_{5,j_2}=1-o(1)$. Thus, by Lemma H.11, we have

    $$\mathbb{E}\left[\left(\mathcal{N}^{(\widetilde{T})}_{s,3,2,i}+\mathcal{N}^{(\widetilde{T})}_{s,3,2,ii}+\mathcal{N}^{(\widetilde{T})}_{s,3,2,iii}\right)\mathbb{1}_{\mathcal{E}_1}\right]\geq\Omega\left(\frac{B}{d}\right),$$

    while

    $$\mathbb{E}\left[\left(\mathcal{N}^{(\widetilde{T})}_{s,4,2,i}+\mathcal{N}^{(\widetilde{T})}_{s,4,2,ii}+\mathcal{N}^{(\widetilde{T})}_{s,4,2,iii}\right)\mathbb{1}_{\mathcal{E}_1}\right]\leq-\Omega\left(\frac{B}{d}\right).$$

  - for $\mathbf{Z}^{2,1}\in\mathcal{E}_2$, $\Lambda^{(\widetilde{T})}_{5,j_2',r_{g_2\cdot y_0}}=(\mathbf{Attn}^{(\widetilde{T})}_{\mathsf{ans},1\to\mathsf{pred},2}+\mathbf{Attn}^{(\widetilde{T})}_{\mathsf{ans},1\to\mathsf{pred},1}-\mathbf{Attn}^{(\widetilde{T})}_{\mathsf{ans},1\to\mathsf{ans},0})2B\leq(0.4+\frac{4}{3}\widetilde{c}-0.4+2\widetilde{c})2B=\frac{5\log d}{6}$. Thus $\mathbf{logit}^{(1)}_{5,j_2}=O(\frac{1}{d^{1/6}})$. Hence by Lemma H.11, we have

    $$\mathbb{E}\left[\left(\mathcal{N}^{(\widetilde{T})}_{s,3,2,i}+\mathcal{N}^{(\widetilde{T})}_{s,3,2,ii}+\mathcal{N}^{(\widetilde{T})}_{s,3,2,iii}\right)\mathbb{1}_{\mathcal{E}_1}\right]\geq-O\left(\frac{B}{d^{7/6}n_y}\right),$$

    while

    $$\mathbb{E}\left[\left(\mathcal{N}^{(\widetilde{T})}_{s,4,2,i}+\mathcal{N}^{(\widetilde{T})}_{s,4,2,ii}+\mathcal{N}^{(\widetilde{T})}_{s,4,2,iii}\right)\mathbb{1}_{\mathcal{E}_1}\right]\leq-\Omega\left(\frac{B}{n_yd}\right).$$

  - $\mathbf{Z}^{2,1}\in\mathcal{E}_3$, we can use the following naive bounds for $p\in\{3,4\}$:

    $$\left|\mathbb{E}\left[\left(\mathcal{N}^{(\widetilde{T})}_{s,p,2,i}+\mathcal{N}^{(\widetilde{T})}_{s,p,2,ii}+\mathcal{N}^{(\widetilde{T})}_{s,p,2,iii}\right)\mathbb{1}_{\mathcal{E}_1}\right]\right|\leq O\left(\frac{B}{dn_y}\right).$$

  Putting them together, combining with the fact that the gradient contributed by $\mathcal{E}_4\cup\mathcal{E}_5\cup\mathcal{E}_6$ is negligible, we can conclude that

  $$\left[-\nabla_{\mathbf{Q}^{(\widetilde{T})}_{4,3}}\mathsf{Loss}^{2,2}_5\right]_{s,s}\geq\Omega(\frac{B}{d}),\ \left[-\nabla_{\mathbf{Q}^{(\widetilde{T})}_{4,4}}\mathsf{Loss}^{2,2}_5\right]_{s,s}\leq-\Omega(\frac{B}{d}),$$

  which implies $\mathbf{Attn}^{(\widetilde{T})}_{\mathsf{ans},1\to\mathsf{ans},1}$ cannot further increase above $0.4-2\widetilde{c}$.

- If $\mathbf{Attn}^{(\widetilde{T})}_{\mathsf{ans},1\to\mathsf{pred},2}-\mathbf{Attn}^{(\widetilde{T})}_{\mathsf{ans},1\to\mathsf{ans},0}<2\varrho$, we shift our focus to the comparison between event $\mathcal{E}_2$ $\mathcal{E}_3$,

  - for $\mathbf{Z}^{2,1}\in\mathcal{E}_1$, we have

    $$\mathbb{E}\left[\left(\mathcal{N}^{(\widetilde{T})}_{s,3,2,i}+\mathcal{N}^{(\widetilde{T})}_{s,3,2,ii}+\mathcal{N}^{(\widetilde{T})}_{s,3,2,iii}\right)\mathbb{1}_{\mathcal{E}_1}\right]\geq-\widetilde{O}(\sigma_0^q),$$

    while

    $$\mathbb{E}\left[\left(\mathcal{N}^{(\widetilde{T})}_{s,4,2,i}+\mathcal{N}^{(\widetilde{T})}_{s,4,2,ii}+\mathcal{N}^{(\widetilde{T})}_{s,4,2,iii}\right)\mathbb{1}_{\mathcal{E}_1}\right]\leq\widetilde{O}(\sigma_0^q).$$

  - for $\mathbf{Z}^{2,1}\in\mathcal{E}_2$, similarly as previous case, we have

    $$\mathbb{E}\left[\left(\mathcal{N}^{(\widetilde{T})}_{s,3,2,i}+\mathcal{N}^{(\widetilde{T})}_{s,3,2,ii}+\mathcal{N}^{(\widetilde{T})}_{s,3,2,iii}\right)\mathbb{1}_{\mathcal{E}_2}\right]\geq-O\left(\frac{B}{d^{7/6}n_y}\right),$$

while
$$\mathbb{E}\left[\left(\mathcal{N}_{s,4,2,i}^{(\widetilde{T})} + \mathcal{N}_{s,4,2,ii}^{(\widetilde{T})} + \mathcal{N}_{s,4,2,iii}^{(\widetilde{T})}\right)\mathbb{1}_{\mathcal{E}_2}\right] \leq O\left(\frac{B}{d^{7/6}n_y}\right).$$

- $\mathbf{Z}^{2,1} \in \mathcal{E}_3$, $\left|\Lambda_{5,j_2',r_{g_1 \cdot y_1}}^{(\widetilde{T})} - \Lambda_{5,j_2,r_{g_2 \cdot y_1}}^{(\widetilde{T})}\right| \leq 4\rho B = o(1)$, hence, we have

$$\mathbb{E}\left[\left(\mathcal{N}_{s,3,2,i}^{(\widetilde{T})} + \mathcal{N}_{s,3,2,ii}^{(\widetilde{T})} + \mathcal{N}_{s,3,2,iii}^{(\widetilde{T})}\right)\mathbb{1}_{\mathcal{E}_3}\right] \geq \Omega\left(\frac{B}{dn_y}\right),$$

while
$$\mathbb{E}\left[\left(\mathcal{N}_{s,4,2,i}^{(\widetilde{T})} + \mathcal{N}_{s,4,2,ii}^{(\widetilde{T})} + \mathcal{N}_{s,4,2,iii}^{(\widetilde{T})}\right)\mathbb{1}_{\mathcal{E}_3}\right] \leq O\left(\frac{B\log\log d}{dn_y \cdot \log d}\right).$$

Putting them together, again we can conclude that

$$\left[-\nabla_{\mathbf{Q}_{4,3}^{(\widetilde{T})}}\mathsf{Loss}_5^{2,2}\right]_{s,s} \geq \Omega\left(\frac{B}{n_y d}\right), \quad \left[-\nabla_{\mathbf{Q}_{4,4}^{(\widetilde{T})}}\mathsf{Loss}_5^{2,2}\right]_{s,s} \leq O\left(\frac{B\log\log d}{dn_y \cdot \log d}\right),$$

which implies $\mathbf{Attn}_{\mathsf{ans},1\to\mathsf{ans},1}^{(\widetilde{T})}$ cannot further increase above $0.4 - 2\widetilde{c}$.

Consequently, this leads to a contradiction, and $\mathbf{Attn}_{\mathsf{ans},1\to\mathsf{ans},1}^{(T_{1,2,1,s})} < 0.4 - \widetilde{c}$, where $\widetilde{c} = \frac{\log d}{8B}$. Then it would follow that $\mathbf{Attn}_{\mathsf{ans},1\to\mathsf{pred},2}^{(T_{1,2,1,s})} \geq \widetilde{c}$, and thus $[\mathbf{Q}_{4,3}^{(T_{1,2,1,s})}]_{s,s} \geq \Omega(1)$. $\qquad\square$

## H.4  Stage 1.2.2: Convergence with Small Wrong Attention

Recall that $\mathsf{Loss}_{5,s}^{2,2} = -\mathbb{E}\left[\log p_F(\mathbf{Z}_{\mathsf{ans},2,5}|\mathbf{Z}^{(2,1)})\big|\tau(x_1) = s\right]$ for $s \in \tau(\mathcal{X})$.

**Induction H.2.** *Given $s \in \tau(\mathcal{X})$, let $T_{1,2,2,s}$ denote the first time that $\mathsf{Loss}_{5,s}^{2,2}$ decreases below $\Theta\left(e^{(-\frac{1}{2}+3.01c_1)\cdot 2B}\right)$. For all iterations $T_{1,2,1,s} \leq t < T_{1,2,2,s}$, we have the following holds*

(a) $\left[\mathbf{Q}_{4,3}^{(t)}\right]_{s,s} + \left[\mathbf{Q}_{4,4}^{(t)}\right]_{s,s}$ *monotonically increases;*

(b) *for $p \in \{3,4\}$, for $j \in \tau(\mathcal{X}) \neq s$, $|[\mathbf{Q}_{4,p}^{(t)}]_{s,j}| \leq O(\frac{[\mathbf{Q}_{4,p}^{(t)}]_{s,j}}{d})$ ;*

(c) *for any sample $\mathbf{Z}^{2,1}$, we have $\mathbf{Attn}_{\mathsf{ans},1\to\mathsf{pred},2}^{(t)} - \mathbf{Attn}_{\mathsf{ans},1\to\mathsf{ans},1}^{(t)} \leq c_1$ for some sufficiently small constant $c_1 = \frac{1.005\log d}{B} > 0$;*

(d) *for any $\mathbf{Z}^{2,1}$, we have $\mathbf{Attn}_{\mathsf{ans},1\to\mathsf{ans},1}^{(t)} - \mathbf{Attn}_{\mathsf{ans},1\to\mathsf{pred},2}^{(t)} \leq \min\left\{\mathbf{Attn}_{\mathsf{ans},1\to\mathsf{pred},1}^{(t)} - c_2, 0\right\}$, where $c_2 = \frac{\log d}{4B} > 0$ is some sufficiently small constant.*

### H.4.1  Attention and Lambda Preliminaries

**Lemma H.24.** *If Induction H.2 holds for all iterations $[T_{1,2,1,s}, t)$,then for any sample $\mathbf{Z}^{2,1}$, we have*

1. $\mathbf{Attn}_{\mathsf{ans},1\to\mathsf{pred},1}^{(t)} + \mathbf{Attn}_{\mathsf{ans},1\to\mathsf{ans},1}^{(t)} \in \left[\frac{4}{3}c_1, 0.4 \pm \widetilde{O}(\frac{1}{d})\right];$

2. $\mathbf{Attn}_{\mathsf{ans},1\to\mathsf{pred},2}^{(t)} - \mathbf{Attn}_{\mathsf{ans},1\to\mathsf{ans},0}^{(t)} \geq \Omega(1);$

3. $\left|\mathbf{Attn}_{\mathsf{ans},1\to\mathsf{pred},1}^{(t)} - \mathbf{Attn}_{\mathsf{ans},1\to\mathsf{ans},0}^{(t)}\right| \leq \widetilde{O}(\frac{1}{d}).$

*Proof.* In the following, we focus on the main events $\mathcal{E}_1$, $\mathcal{E}_2$, and $\mathcal{E}_3$, which correspond to cases where some confused wrong predictions occur. We denote

$$\mathbf{Attn}_{\mathsf{ans},1\to\mathsf{pred},1}^{(t)}, \quad \mathbf{Attn}_{\mathsf{ans},1\to\mathsf{ans},1}^{(t)} = c + \widetilde{O}\left(\frac{1}{d}\right).$$

- If $\mathbf{Attn}_{\mathsf{ans},1\to\mathsf{pred},2}^{(t)} \geq \mathbf{Attn}_{\mathsf{ans},1\to\mathsf{ans},1}^{(t)}$, then

- If $\mathbf{Z}^{2,1} \in \mathcal{E}_1$, we have

$$\log p_F(\mathbf{Z}_{\mathsf{ans},2,5}|\mathbf{Z}^{(2,1)})$$

$$\leq \Theta(1) \cdot \max\left\{ e^{\left(\mathbf{Attn}^{(t)}_{\mathsf{ans},1\to\mathsf{pred},2} - \mathbf{Attn}^{(t)}_{\mathsf{ans},1\to\mathsf{ans},1}\right)2B - \left(\mathbf{Attn}^{(t)}_{\mathsf{ans},1\to\mathsf{pred},2} - \mathbf{Attn}^{(t)}_{\mathsf{ans},1\to\mathsf{ans},0}\right)2B}, \right.$$

$$\left. e^{\log d - \left(\mathbf{Attn}^{(t)}_{\mathsf{ans},1\to\mathsf{pred},2} - \mathbf{Attn}^{(t)}_{\mathsf{ans},1\to\mathsf{ans},0}\right)2B} \right\}$$

$$\leq \Theta\left( e^{2c_1 B - \left(\mathbf{Attn}^{(t)}_{\mathsf{ans},1\to\mathsf{pred},2} - \mathbf{Attn}^{(t)}_{\mathsf{ans},1\to\mathsf{ans},0}\right)2B} \right)$$

$$\leq \Theta\left( e^{-2\left(\frac{1}{2} - 2c - c_1\right)B} \right),$$

where the last inequality follows from the fact that $\mathbf{Attn}^{(t)}_{\mathsf{ans},1\to\mathsf{pred},2} \geq \frac{1}{2}(1 - 2c) = \frac{1}{2} - c$.

- If $\mathbf{Z}^{2,1} \in \mathcal{E}_2$, we have

$$\log p_F(\mathbf{Z}_{\mathsf{ans},2,5}|\mathbf{Z}^{(2,1)}) \leq \Theta(1)$$

$$\cdot \max\left\{ e^{\left(\mathbf{Attn}^{(t)}_{\mathsf{ans},1\to\mathsf{pred},2} + \mathbf{Attn}^{(t)}_{\mathsf{ans},1\to\mathsf{pred},1} - \mathbf{Attn}^{(t)}_{\mathsf{ans},1\to\mathsf{ans},1}\right)B - \left(\mathbf{Attn}^{(t)}_{\mathsf{ans},1\to\mathsf{pred},2} - \mathbf{Attn}^{(t)}_{\mathsf{ans},1\to\mathsf{ans},0}\right)2B}, \right.$$

$$\left. e^{\log d - \left(\mathbf{Attn}^{(t)}_{\mathsf{ans},1\to\mathsf{pred},2} - \mathbf{Attn}^{(t)}_{\mathsf{ans},1\to\mathsf{ans},0}\right)2B} \right\}$$

$$\leq \Theta\left( e^{2(c+c_1)B - \left(\mathbf{Attn}^{(t)}_{\mathsf{ans},1\to\mathsf{pred},2} - \mathbf{Attn}^{(t)}_{\mathsf{ans},1\to\mathsf{ans},0}\right)2B} \right)$$

$$\leq \Theta\left( e^{-2\left(\frac{1}{2} - 3c - c_1\right)B} \right).$$

- If $\mathbf{Z}^{2,1} \in \mathcal{E}_3$, we have

$$\log p_F(\mathbf{Z}_{\mathsf{ans},2,5}|\mathbf{Z}^{(2,1)})$$

$$\leq \Theta(1) \cdot \max\left\{ e^{\left(\mathbf{Attn}^{(t)}_{\mathsf{ans},1\to\mathsf{pred},1} - \mathbf{Attn}^{(t)}_{\mathsf{ans},1\to\mathsf{pred},2}\right)2B}, e^{\log d - \mathbf{Attn}^{(t)}_{\mathsf{ans},1\to\mathsf{pred},2}2B} \right\}$$

$$\leq \Theta(1) \cdot \max\left\{ e^{-2(\frac{1}{2} - c_1 - c)B}, \ e^{-2(\frac{1}{2} - 2c)B} \right\}.$$

- If $\mathbf{Attn}^{(t)}_{\mathsf{ans},1\to\mathsf{pred},2} \leq \mathbf{Attn}^{(t)}_{\mathsf{ans},1\to\mathsf{ans},1}$, then
  - If $\mathbf{Z}^{2,1} \in \mathcal{E}_1$, we have

$$\log p_F(\mathbf{Z}_{\mathsf{ans},2,5}|\mathbf{Z}^{(2,1)}) \leq \Theta(1) \cdot e^{\log d - \left(\mathbf{Attn}^{(t)}_{\mathsf{ans},1\to\mathsf{pred},2} - \mathbf{Attn}^{(t)}_{\mathsf{ans},1\to\mathsf{ans},0}\right)2B}$$

$$\leq \Theta\left( e^{2c_1 B - \left(\mathbf{Attn}^{(t)}_{\mathsf{ans},1\to\mathsf{pred},2} - \mathbf{Attn}^{(t)}_{\mathsf{ans},1\to\mathsf{ans},0}\right)2B} \right)$$

$$\leq \Theta\left( e^{-2\left(\frac{1}{2} - \frac{5}{2}c - \frac{c_2}{2} - c_1\right)B} \right),$$

where the last inequality uses the fact that $\mathbf{Attn}^{(t)}_{\mathsf{ans},1\to\mathsf{ans},1} - \mathbf{Attn}^{(t)}_{\mathsf{ans},1\to\mathsf{pred},2} \leq c - c_2$, implying $\mathbf{Attn}^{(t)}_{\mathsf{ans},1\to\mathsf{pred},2} \geq \frac{1}{2} - \frac{3}{2}c + \frac{c_2}{2}$.

  - If $\mathbf{Z}^{2,1} \in \mathcal{E}_2$, we have

$$\log p_F(\mathbf{Z}_{\mathsf{ans},2,5}|\mathbf{Z}^{(2,1)}) \leq \Theta(1)\cdot$$

$$\max\left\{e^{\left(\mathbf{Attn}^{(t)}_{\mathsf{ans},1\to\mathsf{pred},2}+\mathbf{Attn}^{(t)}_{\mathsf{ans},1\to\mathsf{pred},1}-\mathbf{Attn}^{(t)}_{\mathsf{ans},1\to\mathsf{ans},1}\right)2B-\left(\mathbf{Attn}^{(t)}_{\mathsf{ans},1\to\mathsf{pred},2}-\mathbf{Attn}^{(t)}_{\mathsf{ans},1\to\mathsf{ans},0}\right)2B},\right.$$

$$\left.e^{\log d-\left(\mathbf{Attn}^{(t)}_{\mathsf{ans},1\to\mathsf{pred},2}-\mathbf{Attn}^{(t)}_{\mathsf{ans},1\to\mathsf{ans},0}\right)2B}\right\}$$

$$\leq \Theta\left(e^{\max\{c,c_1\}2B-\left(\mathbf{Attn}^{(t)}_{\mathsf{ans},1\to\mathsf{pred},2}-\mathbf{Attn}^{(t)}_{\mathsf{ans},1\to\mathsf{ans},0}\right)2B}\right)$$

$$\leq \Theta\left(e^{-2\left(\frac{1}{2}-2c-\max\{c,c_1\}\right)B}\right).$$

- If $\mathbf{Z}^{2,1} \in \mathcal{E}_3$, we have

$$\log p_F(\mathbf{Z}_{\mathsf{ans},2,5}|\mathbf{Z}^{(2,1)})$$

$$\leq \Theta(1)\cdot\max\left\{e^{\left(\mathbf{Attn}^{(t)}_{\mathsf{ans},1\to\mathsf{pred},1}-\mathbf{Attn}^{(t)}_{\mathsf{ans},1\to\mathsf{pred},2}\right)2B}, e^{\log d-\mathbf{Attn}^{(t)}_{\mathsf{ans},1\to\mathsf{pred},2}2B}\right\}$$

$$\leq \Theta(1)\cdot\max\left\{e^{-2(\frac{1}{2}-c_1-c)B}, e^{-2(\frac{1}{2}-2c)B}\right\}.$$

Putting all cases together, if $c \leq \frac{2}{3}c_1$, we must have

$$\mathsf{Loss}^{2,2}_{5,s} \leq \Theta\left(e^{-2(\frac{1}{2}+3\cdot\frac{2c_1}{3}+c_1)B}\right) = \Theta\left(e^{(-\frac{1}{2}+3c_1)2B}\right),$$

which contradicts the definition of $T_{1,2,2,s}$. Therefore, it must hold that $c \geq \frac{2}{3}c_1$ for all $t \leq T_{1,2,2,s}$.

$\square$

**Lemma H.25.** *If Induction H.2 holds for all iterations $[T_{1,2,1,s}, t)$, then given $\mathbf{Z}^{2,1} \in \mathcal{E}_1$,*

1. *for the prediction $j_2$, we have*

$$\Lambda^{(t)}_{5,j_2,r_{g_2\cdot y_1}} = \left(\mathbf{Attn}^{(t)}_{\mathsf{ans},1\to\mathsf{pred},2} - \mathbf{Attn}^{(t)}_{\mathsf{ans},1\to\mathsf{ans},0}\right)\cdot 2B + O\left(\frac{B}{n_y}\right) + O(\delta).$$

2. *for the prediction $j'_2 = \tau(g_2(y_0))$, we have*

$$\Lambda^{(t)}_{5,j'_2,r_{g_2\cdot y_0}} = \left(\mathbf{Attn}^{(t)}_{\mathsf{ans},1\to\mathsf{pred},2} - \mathbf{Attn}^{(t)}_{\mathsf{ans},1\to\mathsf{ans},1}\right)\cdot 2B + \widetilde{O}\left(\frac{B}{d\cdot n_y}\right) + O(\delta).$$

3. *for the prediction $\tau(g_1(y_0))$, $r_{g_1\cdot y_0}$ cannot be activated.*

4. *for the prediction $\tau(g_1(y_1))$, we have*

$$\Lambda^{(t)}_{5,\tau(g_1(y_1)),r_{g_1\cdot y_1}} = \left(\mathbf{Attn}^{(t)}_{\mathsf{ans},1\to\mathsf{ans},1} - \mathbf{Attn}^{(t)}_{\mathsf{ans},1\to\mathsf{pred},2}\right)\cdot\frac{2B}{n_y} + \widetilde{O}\left(\frac{B}{d}\right) + O(\delta).$$

5. *for other $j \in \tau(\mathcal{Y})$, if there are $j$ and $y$, s.t., $g_1, g_2 \in \mathrm{Fiber}_{j,y}$ (notice that $y \neq y_0, y_1$ for $\mathcal{E}_1$), then for such a $j$, $r \in \widehat{\mathfrak{A}}_j \setminus \{r_{g_2\cdot y}\}$ cannot be activated, moreover, we have*

$$\Lambda^{(t)}_{5,j,r_{g_2\cdot y}} = \left(\mathbf{Attn}^{(t)}_{\mathsf{ans},1\to\mathsf{pred},2} - \mathbf{Attn}^{(t)}_{\mathsf{ans},1\to\mathsf{ans},1}\right)\cdot 2B + \widetilde{O}\left(\frac{B}{d\cdot n_y}\right) + O(\delta);$$

*else, none of $r \in \widehat{\mathfrak{A}}_j$ can be activated.*

*Notice that for $\mathcal{E}_1$, the neurons mentioned in 1-4 should be different neurons, while the predictions in 1 and 3, or 2 and 4 can be the same in some cases, e.g., $g_1(y_1) = g_2(y_0)$. In all cases, except for the neurons mentioned above, i.e., $\cup_{\ell,\ell'\in[2]}\{r_{g_\ell\cdot y_{\ell'-1}}\}$ (which may not be activated), all other neurons $r \in \cup_{\ell,\ell'\in[2]}\left(\widehat{\mathfrak{A}}_{\tau(g_\ell\cdot y_{\ell'-1})} \setminus \{r_{g_\ell\cdot y_{\ell'-1}}\}\right)$ cannot be activated.*

**Lemma H.26.** *If Induction H.2 holds for all iterations $t \in [T_{1,2,1,s}, T_{1,2,2,s})$, then given $\mathbf{Z}^{2,1} \in \mathcal{E}_2$, we have*

1. *for the prediction $j_2$, $r \in \widehat{\mathfrak{A}}_{j_2} \setminus \{r_{g_2 \cdot y_1}\}$ cannot be activated, moreover, we have*

$$\Lambda^{(t)}_{5,j_2,r_{g_2 \cdot y_1}} = \left(\mathbf{Attn}^{(t)}_{\mathsf{ans},1 \to \mathsf{pred},2} - \mathbf{Attn}^{(t)}_{\mathsf{ans},1 \to \mathsf{ans},0}\right) \cdot 2B + \widetilde{O}\left(\frac{B}{n_y}\right) + O(\delta).$$

2. *for the prediction $j_2' = \tau(g_2(y_0))$, $r \in \widehat{\mathfrak{A}}_{j_2'} \setminus \{r_{g_2 \cdot y_0}\}$ (notice that in this case $r_{g_2 \cdot y_0} = r_{g_1 \cdot y_0}$) cannot be activated, moreover, we have*

$$\Lambda^{(t)}_{5,j_2',r_{g_2 \cdot y_0}} = \left(\mathbf{Attn}^{(t)}_{\mathsf{ans},1 \to \mathsf{pred},2} + \mathbf{Attn}^{(t)}_{\mathsf{ans},1 \to \mathsf{pred},1} - \mathbf{Attn}^{(t)}_{\mathsf{ans},1 \to \mathsf{ans},1}\right) \cdot 2B + O(\delta).$$

3. *for the prediction $\tau(g_1(y_1))$, $r \in \widehat{\mathfrak{A}}_{\tau(g_1(y_1))} \setminus \{r_{g_1 \cdot y_1}\}$ cannot be activated, moreover, we have*

$$\Lambda^{(t)}_{5,\tau(g_1(y_1)),r_{g_1 \cdot y_1}} = \left(\mathbf{Attn}^{(t)}_{\mathsf{ans},1 \to \mathsf{ans},1} - \mathbf{Attn}^{(t)}_{\mathsf{ans},1 \to \mathsf{pred},2}\right) \cdot \frac{2B}{n_y} + O(\delta).$$

4. *for other $j \in \tau(\mathcal{Y})$, if there are $j$ and $y$, s.t., $g_1, g_2 \in \mathrm{Fiber}_{j,y}$ (notice that $y \neq y_0, y_1$ for $\mathcal{E}_2$), then for such a $j$, $r \in \widehat{\mathfrak{A}}_j \setminus \{r_{g_2 \cdot y}\}$ cannot be activated, moreover, we have*

$$\Lambda^{(t)}_{5,j,r_{g_2 \cdot y}} = \left(\mathbf{Attn}^{(t)}_{\mathsf{ans},1 \to \mathsf{pred},2} - \mathbf{Attn}^{(t)}_{\mathsf{ans},1 \to \mathsf{ans},1}\right) \cdot 2B + \widetilde{O}\left(\frac{B}{d \cdot n_y}\right) + O(\delta);$$

*else, none of $r \in \widehat{\mathfrak{A}}_j$ can be activated.*

**Lemma H.27.** *If Induction H.2 holds for all iterations $t \in [T_{1,2,1,s}, T_{1,2,2,s})$, then given $\mathbf{Z}^{2,1} \in \mathcal{E}_3$, we have*

1. *for the prediction $j_2$, $r \in \widehat{\mathfrak{A}}_{j_2} \setminus \{r_{g_2 \cdot y_1}\}$ cannot be activated, moreover, we have*

$$\Lambda^{(t)}_{5,j_2,r_{g_2 \cdot y_1}} = \mathbf{Attn}^{(t)}_{\mathsf{ans},1 \to \mathsf{pred},2} \cdot 2B \pm O(\delta).$$

2. *for the prediction $\tau(g_1(y_1))$, $r \in \widehat{\mathfrak{A}}_{\tau(g_1(y_1))} \setminus \{r_{g_1 \cdot y_1}\}$ cannot be activated, moreover, we have*

$$\Lambda^{(t)}_{5,\tau(g_1(y_1)),r_{g_1 \cdot y_1}} = \mathbf{Attn}^{(t)}_{\mathsf{ans},1 \to \mathsf{pred},1} \cdot 2B \pm O(\delta).$$

3. *for other $j = g(y_1)$, where $g_1, g_2 \notin \mathrm{Fiber}_{j,y_1}$, we have*

$$\Lambda^{(t)}_{5,j,r_{g \cdot y_1}} = \left(\mathbf{Attn}^{(t)}_{\mathsf{ans},1 \to \mathsf{ans},1} - \mathbf{Attn}^{(t)}_{\mathsf{ans},1 \to \mathsf{pred},2}\right) \cdot \frac{2B}{n_y} + \widetilde{O}\left(\frac{B}{d \cdot n_y}\right) + O(\delta);$$

*moreover, if there exists $y$, s.t., $g_1, g_2 \in \mathrm{Fiber}_{j,y}$ (notice that $y \neq y_1$ for $\mathcal{E}_3$), we have*

$$\Lambda^{(t)}_{5,j,r_{g_2 \cdot y}} = \left(\mathbf{Attn}^{(t)}_{\mathsf{ans},1 \to \mathsf{pred},2} - \mathbf{Attn}^{(t)}_{\mathsf{ans},1 \to \mathsf{ans},1}\right) \cdot 2B + \widetilde{O}\left(\frac{B}{d \cdot n_y}\right) + O(\delta);$$

*besides, other $r \in \widehat{\mathfrak{A}}_j$ cannot be activated.*

**Lemma H.28.** *If Induction H.2 holds for all iterations $t \in [T_{1,2,1,s}, T_{1,2,2,s})$, then given $\mathbf{Z}^{2,1} \in \mathcal{E}_4$, we have*

1. *for the prediction $j_2$, $r \in \widehat{\mathfrak{A}}_{j_2} \setminus \{r_{g_2 \cdot y_1}\}$ cannot be activated, moreover, we have*

$$\Lambda^{(t)}_{5,j_2,r_{g_2 \cdot y_1}} = \mathbf{Attn}^{(t)}_{\mathsf{ans},1 \to \mathsf{pred},2} \cdot 2B + O(\delta).$$

2. *for the prediction $\tau(g_2(y_0))$, $r \in \widehat{\mathfrak{A}}_{\tau(g_2(y_0))} \setminus \{r_{g_2 \cdot y_0}\}$ cannot be activated, moreover, we have*

$$\Lambda^{(t)}_{5,j_2',r_{g_2 \cdot y_0}} = \left(\mathbf{Attn}^{(t)}_{\mathsf{ans},1 \to \mathsf{pred},2} + \mathbf{Attn}^{(t)}_{\mathsf{ans},1 \to \mathsf{pred},1} - \mathbf{Attn}^{(t)}_{\mathsf{ans},1 \to \mathsf{ans},1}\right) \cdot 2B + O(\delta).$$

3. *for other $j \in \tau(\mathcal{Y})$, if there are $j$ and $y$, s.t., $g_1, g_2 \in \mathrm{Fiber}_{j,y}$, then for such a $j$, $r \in \widehat{\mathfrak{A}}_j \setminus \{r_{g_2 \cdot y}\}$ cannot be activated, moreover, we have*

$$\Lambda^{(t)}_{5,j,r_{g_2 \cdot y}} = \left(\mathbf{Attn}^{(t)}_{\mathrm{ans},1 \to \mathrm{pred},2} - \mathbf{Attn}^{(t)}_{\mathrm{ans},1 \to \mathrm{ans},1}\right) \cdot 2B + \widetilde{O}\left(\frac{B}{d \cdot n_y}\right) + O(\delta);$$

*else, none of $r \in \widehat{\mathfrak{A}}_j$ can be activated.*

**Lemma H.29.** *If Induction H.2 holds for all iterations $t \in [T_{1,2,1,s}, T_{1,2,2,s})$, then given $\mathbf{Z}^{2,1} \in \mathcal{E}_5$, we have*

1. *for the prediction $j_2$, $r \in \widehat{\mathfrak{A}}_{j_2} \setminus \{r_{g_2 \cdot y_1}\}$ cannot be activated, moreover, we have*

$$\Lambda^{(t)}_{5,j_2,r_{g_2 \cdot y_1}} = \mathbf{Attn}^{(t)}_{\mathrm{ans},1 \to \mathrm{pred},2} \cdot 2B \pm O(\delta).$$

2. *for the prediction $\tau(g_2(y_0))$, $r \in \widehat{\mathfrak{A}}_{\tau(g_2(y_0))} \setminus \{r_{g_2 \cdot y_0}\}$ cannot be activated, moreover, we have*

$$\Lambda^{(t)}_{5,j_2',r_{g_2 \cdot y_0}} = \left(\mathbf{Attn}^{(t)}_{\mathrm{ans},1 \to \mathrm{pred},2} - \mathbf{Attn}^{(t)}_{\mathrm{ans},1 \to \mathrm{ans},1}\right) \cdot 2B + \widetilde{O}\left(\frac{B}{d \cdot n_y}\right) + O(\delta).$$

3. *for the prediction $\tau(g_1(y_0))$, none of $r \in \widehat{\mathfrak{A}}_{\tau(g_2(y_0))}$ can be activated.*

4. *for other $j \in \tau(\mathcal{Y})$, if there are $j$ and $y$, s.t., $g_1, g_2 \in \mathrm{Fiber}_{j,y}$, then for such a $j$, $r \in \widehat{\mathfrak{A}}_j \setminus \{r_{g_2 \cdot y}\}$ cannot be activated, moreover, we have*

$$\Lambda^{(t)}_{5,j,r_{g_2 \cdot y}} = \left(\mathbf{Attn}^{(t)}_{\mathrm{ans},1 \to \mathrm{pred},2} - \mathbf{Attn}^{(t)}_{\mathrm{ans},1 \to \mathrm{ans},1}\right) \cdot 2B + \widetilde{O}\left(\frac{B}{d \cdot n_y}\right) + O(\delta);$$

*else, none of $r \in \widehat{\mathfrak{A}}_j$ can be activated.*

**Lemma H.30.** *If Induction H.2 holds for all iterations $t \in [T_{1,2,1,s}, T_{1,2,2,s})$, then given $\mathbf{Z}^{2,1} \in \mathcal{E}_6$, we have*

1. *for the prediction $j_2$, $r \in \widehat{\mathfrak{A}}_{j_2} \setminus \{r_{g_2 \cdot y_1}\}$ cannot be activated, moreover, we have*

$$\Lambda^{(t)}_{5,j_2,r_{g_2 \cdot y_1}} = \left(\mathbf{Attn}^{(t)}_{\mathrm{ans},1 \to \mathrm{pred},2} + \mathbf{Attn}^{(t)}_{\mathrm{ans},1 \to \mathrm{pred},1}\right) \cdot 2B \pm O(\delta).$$

2. *for other $j = g(y_1)$, where $g_2 \notin \mathrm{Fiber}_{j,y_1}$, we have $r_{g \cdot y_1}$ cannot be activated, moreover, if there exists $y \neq y_1$, s.t., $g_1, g_2 \in \mathrm{Fiber}_{j,y}$, we have*

$$\Lambda^{(t)}_{5,j,r_{g_2 \cdot y}} = \left(\mathbf{Attn}^{(t)}_{\mathrm{ans},1 \to \mathrm{pred},2} - \mathbf{Attn}^{(t)}_{\mathrm{ans},1 \to \mathrm{ans},1}\right) \cdot 2B + \widetilde{O}\left(\frac{B}{d \cdot n_y}\right) + O(\delta);$$

*besides, other $r \in \widehat{\mathfrak{A}}_j$ cannot be activated.*

### H.4.2 Gradient Lemma

**Lemma H.31.** *If Induction H.2 holds for all iterations $t \in [T_{1,2,1,s}, T_{1,2,2,s})$, given $s \in \tau(\mathcal{X})$, if $\mathbb{E}\left[\mathbf{Attn}^{(t)}_{\mathrm{ans},1 \to \mathrm{pred},2} \mid \tau(x_1) = s\right] \leq \frac{1}{2}$ we have*

$$\left[-\nabla_{\mathbf{Q}^{(t)}_{4,3}} \mathsf{Loss}^{2,2}_5\right]_{s,s} + \left[-\nabla_{\mathbf{Q}^{(t)}_{4,4}} \mathsf{Loss}^{2,2}_5\right]_{s,s}$$

$$\geq \Omega\left(\frac{B}{d}\right) \cdot \mathbb{E}\left[(1 - \mathbf{logit}^{(t)}_{5,j_2})|\tau(x_1) = s, \mathcal{E}_1\right] + \Omega\left(\frac{B}{n_y d}\right) \cdot \sum_{m=2}^{3} \mathbb{E}\left[(1 - \mathbf{logit}^{(t)}_{5,j_2})|\tau(x_1) = s, \mathcal{E}_m\right].$$

*Proof.* The analysis follows the similar idea as Lemma H.19, while Induction H.2 ensures that $\mathbb{E}\left[\mathbf{Attn}^{(t)}_{\mathrm{ans},1 \to \mathrm{pred},2} - \mathbf{Attn}^{(t)}_{\mathrm{ans},1 \to \mathrm{ans},0} \mid \tau(x_1) = s\right] \geq \Omega(1) > \frac{\varrho}{B}$, which falls into the regime (ii). Thus for the main event $\mathcal{E}_1$, the term $\Lambda^{(t)}5, j_2, rg_2 \cdot y_1$ is guaranteed to remain within the linear regime. We can first directly upper bound the term $\mathcal{N}^{(t)}_{s,2,iii}$ by $\widetilde{O}(\sigma_0^q)$.

(a) For $\mathbf{Z}^{2,1} \in \mathcal{E}_1$, by Induction H.2 and Lemma H.25, we can obtain

$$\mathbb{E}\left[\mathcal{N}_{s,2,i}^{(t)}\mathbb{1}_{\mathcal{E}_1}\right]$$

$$= \mathbb{E}\left[(1 - \mathbf{logit}_{5,j_2}^{(t)}) \cdot \left(\left(-\mathbf{Attn}_{\mathsf{ans},1\to\mathsf{ans},0}^{(t)} \cdot V_{j_2,r_{g_2\cdot y_1}}(y_0) - \mathbf{Attn}_{\mathsf{ans},1\to\mathsf{pred},1}^{(t)} \cdot V_{j_2,r_{g_2\cdot y_1}}(g_1)\right.\right.\right.$$

$$\left.\left.\left.+ \left(1 - \mathbf{Attn}_{\mathsf{ans},1\to\mathsf{ans},1}^{(t)} - \mathbf{Attn}_{\mathsf{ans},1\to\mathsf{pred},2}^{(t)}\right)\Lambda_{5,j_2,r_{g_2\cdot y_1}}^{(t)} \pm \widetilde{O}(\sigma_0)\right) \pm \widetilde{O}(\delta^q)\right)\mathbb{1}_{\tau(x_1)=s}\mathbb{1}_{\mathcal{E}_1}\right]$$

$$\tag{114}$$

$$\geq \Omega\left(\frac{B}{d}\right) \cdot \mathbb{E}\left[(1 - \mathbf{logit}_{5,j_2}^{(t)})\big|\tau(x_1) = s, \mathcal{E}_1\right].$$

Moving to $\mathcal{N}_{s,2,ii}^{(t)}$, by Lemma H.25, we have

- for $j = \tau(g_1(y_1))$, $r = r_{g_1\cdot y_1}$, due to the cancellation of $\mathbf{Attn}_{\mathsf{ans},1\to\mathsf{ans},0}^{(t)} \cdot V_{j,r}(y_0) + \mathbf{Attn}_{\mathsf{ans},1\to\mathsf{pred},1}^{(t)} \cdot V_{j,r}(g_1)$, we have

$$\left(\mathbf{Attn}_{\mathsf{ans},1\to\mathsf{ans},0}^{(t)} \cdot V_{j,r}(y_0) + \mathbf{Attn}_{\mathsf{ans},1\to\mathsf{pred},1}^{(t)} \cdot V_{j,r}(g_1)\right.$$

$$\left.- \left(1 - \mathbf{Attn}_{\mathsf{ans},1\to\mathsf{ans},1}^{(t)} - \mathbf{Attn}_{\mathsf{ans},1\to\mathsf{pred},2}^{(t)}\right)\Lambda_{5,j,r}^{(t)} \pm \widetilde{O}(\sigma_0)\right)$$

$$\geq -\left(1 - \mathbf{Attn}_{\mathsf{ans},1\to\mathsf{ans},1}^{(t)} - \mathbf{Attn}_{\mathsf{ans},1\to\mathsf{pred},2}^{(t)}\right)\Lambda_{5,j,r}^{(t)}.$$

- for $j = \tau(g_2(y_0))$, $r = r_{g_2\cdot y_0}$

$$\left(\mathbf{Attn}_{\mathsf{ans},1\to\mathsf{ans},0}^{(t)} \cdot V_{j,r}(y_0) + \mathbf{Attn}_{\mathsf{ans},1\to\mathsf{pred},1}^{(t)} \cdot V_{j,r}(g_1)\right.$$

$$\left.- \left(1 - \mathbf{Attn}_{\mathsf{ans},1\to\mathsf{ans},1}^{(t)} - \mathbf{Attn}_{\mathsf{ans},1\to\mathsf{pred},2}^{(t)}\right)\Lambda_{5,j,r}^{(t)} \pm \widetilde{O}(\sigma_0)\right)$$

$$\geq -\left(1 - \mathbf{Attn}_{\mathsf{ans},1\to\mathsf{ans},1}^{(t)} - \mathbf{Attn}_{\mathsf{ans},1\to\mathsf{pred},2}^{(t)}\right)\Lambda_{5,j,r}^{(t)}.$$

- for $j = g_1(y)$, where $\exists y \neq y_0, y_1$, s.t., $g_2(y) = g_1(y)$, we have

$$\left(\mathbf{Attn}_{\mathsf{ans},1\to\mathsf{ans},0}^{(t)} \cdot V_{j,r}(y_0) + \mathbf{Attn}_{\mathsf{ans},1\to\mathsf{pred},1}^{(t)} \cdot V_{j,r}(g_1)\right.$$

$$\left.- \left(1 - \mathbf{Attn}_{\mathsf{ans},1\to\mathsf{ans},1}^{(t)} - \mathbf{Attn}_{\mathsf{ans},1\to\mathsf{pred},2}^{(t)}\right)\Lambda_{5,j,r}^{(t)} \pm \widetilde{O}(\sigma_0)\right)$$

$$\geq -\left(1 - \mathbf{Attn}_{\mathsf{ans},1\to\mathsf{ans},1}^{(t)} - \mathbf{Attn}_{\mathsf{ans},1\to\mathsf{pred},2}^{(t)}\right)\Lambda_{5,j,r}^{(t)}.$$

Putting them together, and upper bound $\sum_{j\neq j_2\in\tau(\mathcal{Y})} \mathbf{logit}_{5,j}^{(t)}$ by $1 - \mathbf{logit}_{5,j_2}^{(t)}$, we have

$$\mathbb{E}\left[\mathcal{N}_{s,2,ii}^{(t)}\mathbb{1}_{\mathcal{E}_1}\right]$$

$$= \mathbb{E}\left[\sum_{j\neq j_2\in\tau(\mathcal{Y})} \mathbf{logit}_{5,j}^{(t)} \cdot \left(\sum_{r\in\widehat{\mathfrak{A}}_j} \mathbf{sReLU}'(\Lambda_{5,j,r})\cdot\right.\right.$$

$$\left(\mathbf{Attn}_{\mathsf{ans},1\to\mathsf{ans},0}^{(t)} \cdot V_{j,r}(y_0) + \mathbf{Attn}_{\mathsf{ans},1\to\mathsf{pred},1}^{(t)} \cdot V_{j,r}(g_1)\right.$$

$$\left.\left.\left.- \left(1 - \mathbf{Attn}_{\mathsf{ans},1\to\mathsf{ans},1}^{(t)} - \mathbf{Attn}_{\mathsf{ans},1\to\mathsf{pred},2}^{(t)}\right)\Lambda_{5,j,r}^{(t)} \pm \widetilde{O}(\sigma_0)\right) \pm \widetilde{O}(\delta^q)\right)\mathbb{1}_{\tau(x_1)=s}\mathbb{1}_{\mathcal{E}_1}\right]$$

$$\geq -\mathbb{E}\left[\left(1 - \mathbf{logit}_{5,j_2}^{(t)}\right) \cdot \left(1 - \mathbf{Attn}_{\mathsf{ans},1\to\mathsf{ans},1}^{(t)} - \mathbf{Attn}_{\mathsf{ans},1\to\mathsf{pred},2}^{(t)}\right)\right.$$

$$\left.\cdot \max_{y\neq y_0,y_1}\left\{\Lambda_{5,\tau(g_1(y_1)),r_{g_1\cdot y_1}}^{(t)}, \Lambda_{5,\tau(g_2(y_0)),r_{g_2\cdot y_0}}^{(t)}\right\}\mathbb{1}_{\tau(x_1)=s}\mathbb{1}_{\mathcal{E}_1}\right],$$

where the last inequality is due to the fact that $\Lambda^{(t)}_{5,\tau(g_2(y_0)),r_{g_2 \cdot y_0}} = \Lambda^{(t)}_{5,\tau(g_2(y))),r_{g_2 \cdot y}} \pm \widetilde{O}(\frac{B}{d \cdot n_y})$ for $y \neq y_0, y_1$ s.t., $g_1(y) = g_2(y)$. Notice that

$$\max_{y \neq y_0, y_1} \left\{ \Lambda^{(t)}_{5,\tau(g_1(y_1)),r_{g_1 \cdot y_1}}, \Lambda^{(t)}_{5,\tau(g_2(y_0)),r_{g_2 \cdot y_0}} \right\} \leq$$

$$\max \left\{ \mathbf{Attn}^{(t)}_{\mathsf{ans},1 \to \mathsf{pred},2} - \mathbf{Attn}^{(t)}_{\mathsf{ans},1 \to \mathsf{ans},1}, \Theta(\frac{1}{n_y}) \right\} 2B,$$

while

$$(114) \geq \mathbb{E}\left[ \left(1 - \mathbf{logit}^{(t)}_{5,j_2}\right) \cdot \mathbf{Attn}^{(t)}_{\mathsf{ans},1 \to \mathsf{ans},0} \cdot 2B \mathbb{1}_{\tau(x_1)=s} \mathbb{1}_{\mathcal{E}_1} \right]$$

$$= \mathbb{E}\left[ \left(1 - \mathbf{logit}^{(t)}_{5,j_2}\right) \cdot \left(1 - \mathbf{Attn}^{(t)}_{\mathsf{ans},1 \to \mathsf{ans},1} - \mathbf{Attn}^{(t)}_{\mathsf{ans},1 \to \mathsf{pred},2}\right) B \mathbb{1}_{\tau(x_1)=s} \mathbb{1}_{\mathcal{E}_1} \right].$$

Since by Induction H.2, $\mathbf{Attn}^{(t)}_{\mathsf{ans},1 \to \mathsf{pred},2} - \mathbf{Attn}^{(t)}_{\mathsf{ans},1 \to \mathsf{ans},1} \leq c_1 \ll \frac{1}{2}$, thus we have

$$\mathbb{E}\left[ \mathcal{N}^{(t)}_{s,2,i} \mathbb{1}_{\mathcal{E}_1} \right] + \mathbb{E}\left[ \mathcal{N}^{(t)}_{s,2,ii} \mathbb{1}_{\mathcal{E}_1} \right] \geq \Omega\left(\frac{B}{d}\right) \cdot \mathbb{E}\left[ (1 - \mathbf{logit}^{(t)}_{5,j_2}) | \tau(x_1) = s, \mathcal{E}_1 \right].$$

(b) For $\mathbf{Z}^{2,1} \in \mathcal{E}_2$, $\mathcal{N}^{(t)}_{s,2,i}$ can be bounded analogously to (114), yielding

$$\mathbb{E}\left[ \mathcal{N}^{(t)}_{s,2,i} \mathbb{1}_{\mathcal{E}_2} \right] \geq \Omega\left(\frac{B}{d \cdot n_y}\right) \cdot \mathbb{E}\left[ (1 - \mathbf{logit}^{(t)}_{5,j_2}) | \tau(x_1) = s, \mathcal{E}_2 \right] \geq 0.$$

Moving to $\mathcal{N}^{(t)}_{s,2,ii}$,

- For $j = \tau(g_1(y_1))$, we obtain

$$= \mathbb{E}\Big[ \mathbf{logit}^{(t)}_{5,j} \Big( \mathbf{sReLU}'(\Lambda_{5,j,r_{g_1 \cdot y_1}}) \cdot$$

$$\Big( \mathbf{Attn}^{(t)}_{\mathsf{ans},1 \to \mathsf{ans},0} \cdot V_{j,r_{g_1 \cdot y_1}}(y_0) + \mathbf{Attn}^{(t)}_{\mathsf{ans},1 \to \mathsf{pred},1} \cdot V_{j,r_{g_1 \cdot y_1}}(g_1) \tag{115}$$

$$- \left(1 - \mathbf{Attn}^{(t)}_{\mathsf{ans},1 \to \mathsf{ans},1} - \mathbf{Attn}^{(t)}_{\mathsf{ans},1 \to \mathsf{pred},2}\right) \Lambda^{(t)}_{5,j,r} \pm \widetilde{O}(\sigma_0) \Big) \pm \widetilde{O}(\delta^q) \Big) \mathbb{1}_{\tau(x_1)=s} \mathbb{1}_{\mathcal{E}_2} \Big]$$

$$\geq -O\left(\frac{B}{n_y^2 \cdot d}\right) \cdot \mathbb{E}\left[ (1 - \mathbf{logit}^{(t)}_{5,j_2}) | \tau(x_1) = s, \mathcal{E}_2 \right],$$

where the inequality follows from the cancellation of the term

$$\mathbf{Attn}^{(t)}_{\mathsf{ans},1 \to \mathsf{ans},0} \cdot V_{j,r_{g_1 \cdot y_1}}(y_0) + \mathbf{Attn}^{(t)}_{\mathsf{ans},1 \to \mathsf{pred},1} \cdot V_{j,r_{g_1 \cdot y_1}}(g_1),$$

together with the fact that

$$\Lambda^{(t)}_{5,\tau(g_1(y_1)),r_{g_1 \cdot y_1}} = \left( \mathbf{Attn}^{(t)}_{\mathsf{ans},1 \to \mathsf{ans},1} - \mathbf{Attn}^{(t)}_{\mathsf{ans},1 \to \mathsf{pred},2} \right) \cdot \frac{2B}{n_y} + O(\delta) \leq O\left(\frac{B}{n_y}\right).$$

- For $j_2' = \tau(g_2(y_0)) = \tau(g_1(y_0))$, it is clear that

$$\mathbb{E}\Big[ \mathbf{logit}^{(t)}_{5,j_2'} \Big( \mathbf{sReLU}'(\Lambda^{(t)}_{5,j_2',r_{g_1 \cdot y_0}}) \cdot$$

$$\Big( \mathbf{Attn}^{(t)}_{\mathsf{ans},1 \to \mathsf{ans},0} \cdot V_{j_2',r_{g_1 \cdot y_0}}(y_0) + \mathbf{Attn}^{(t)}_{\mathsf{ans},1 \to \mathsf{pred},1} \cdot V_{j_2',r_{g_1 \cdot y_0}}(g_1)$$

$$- \left(1 - \mathbf{Attn}^{(t)}_{\mathsf{ans},1 \to \mathsf{ans},1} - \mathbf{Attn}^{(t)}_{\mathsf{ans},1 \to \mathsf{pred},2}\right) \Lambda^{(t)}_{5,j_2',r_{g_1 \cdot y_0}} \pm \widetilde{O}(\sigma_0) \Big) \pm \widetilde{O}(\delta^q) \Big) \mathbb{1}_{\tau(x_1)=s} \mathbb{1}_{\mathcal{E}_2} \Big]$$

$$\geq 0,$$

where the last inequality is due to the fact that

$$\mathbf{Attn}^{(t)}_{\mathsf{ans},1\to\mathsf{pred},1}\cdot V_{j_2',r_{g_1\cdot y_0}}(g_1)-\big(1-\mathbf{Attn}^{(t)}_{\mathsf{ans},1\to\mathsf{ans},1}-\mathbf{Attn}^{(t)}_{\mathsf{ans},1\to\mathsf{pred},2}\big)\Lambda^{(t)}_{5,j_2',r_{g_1\cdot y_0}}$$

$$\geq\mathbf{Attn}^{(t)}_{\mathsf{ans},1\to\mathsf{pred},1}\bigg(1-2\Big(\mathbf{Attn}^{(t)}_{\mathsf{ans},1\to\mathsf{pred},2}+\mathbf{Attn}^{(t)}_{\mathsf{ans},1\to\mathsf{pred},1}-\mathbf{Attn}^{(t)}_{\mathsf{ans},1\to\mathsf{ans},1}\Big)\bigg)2B$$

$$\geq\mathbf{Attn}^{(t)}_{\mathsf{ans},1\to\mathsf{pred},1}\Big(1-2\big(c_1+0.2\big)\Big)2B\geq0.$$

- For $j=\tau(g_2(y))$ with some $y\neq y_0,y_1$ such that $g_1(y)=g_2(y)$, we distinguish two cases:
  - If
    $$\mathbb{E}\big[\mathbf{Attn}^{(t)}_{\mathsf{ans},1\to\mathsf{ans},1}-\mathbf{Attn}^{(t)}_{\mathsf{ans},1\to\mathsf{pred},2}\mid\tau(x_1)=s\big]\leq\frac{c_1}{4},$$

    then by Induction H.2 and Lemma H.26, we have we have $\Lambda^{(t)}_{5,j_2',r_{g_2\cdot y_0}}-\Lambda^{(t)}_{5,j,r_{g_2\cdot y}}\geq$
    $\big(-\frac{c_1}{6}+\mathbf{Attn}^{(t)}_{\mathsf{ans},1\to\mathsf{pred},1}\big)2B\geq(-\frac{c_1}{4}+\frac{3}{4}c_1)2B=c_1B$, which implies

    $$\sum_{y\neq y_0,y_1}\mathbf{logit}^{(t)}_{5,\tau(g_2(y))}\mathbb{1}_{g_1(y)=g_2(y)}\ll\widetilde{O}\big(\tfrac{1}{d^{1/2}}\big)\cdot\mathbf{logit}^{(t)}_{5,j_2'}.$$

  Consequently,

  $$\sum_{y\neq y_0,y_1}\bigg|\mathbb{E}\bigg[\mathbf{logit}^{(t)}_{5,j}\Big(\mathbf{sReLU}'(\Lambda^{(t)}_{5,j,r_{g_1\cdot y}})\cdot$$

  $$\Big(\mathbf{Attn}^{(t)}_{\mathsf{ans},1\to\mathsf{ans},0}\cdot V_{j,r_{g_1\cdot y}}(y_0)+\mathbf{Attn}^{(t)}_{\mathsf{ans},1\to\mathsf{pred},1}\cdot V_{j,r_{g_1\cdot y}}(g_1)$$

  $$-\big(1-\mathbf{Attn}^{(t)}_{\mathsf{ans},1\to\mathsf{ans},1}-\mathbf{Attn}^{(t)}_{\mathsf{ans},1\to\mathsf{pred},2}\big)\Lambda^{(t)}_{5,j,r_{g_1\cdot y}}\pm\widetilde{O}(\sigma_0)\Big)\pm\widetilde{O}(\delta^q)\Big)$$

  $$\mathbb{1}_{\tau(x_1)=s}\mathbb{1}_{\mathcal{E}_2}\mathbb{1}_{g_1(y)=g_2(y)}\bigg]\bigg|\leq O\big(\tfrac{1}{d^{1/2}}\big)\cdot(115).$$

  - If
    $$\mathbb{E}\big[\mathbf{Attn}^{(t)}_{\mathsf{ans},1\to\mathsf{ans},1}-\mathbf{Attn}^{(t)}_{\mathsf{ans},1\to\mathsf{pred},2}\mid\tau(x_1)=s\big]\geq\frac{c_1}{4},$$

    then by Lemma H.26, the activation $\Lambda^{(t)}_{5,j,r_{g_1\cdot y}}$ cannot be triggered.

Putting the above discussion together, we have

$$\mathbb{E}\Big[\mathcal{N}^{(t)}_{s,2,ii}\mathbb{1}_{\mathcal{E}_1}\Big]+\mathbb{E}\Big[\mathcal{N}^{(t)}_{s,2,ii}\mathbb{1}_{\mathcal{E}_2}\Big]\geq\Omega\Big(\frac{B}{d\cdot n_y}\Big)\cdot\mathbb{E}\Big[(1-\mathbf{logit}^{(t)}_{5,j_2})\big|\tau(x_1)=s,\mathcal{E}_2\Big].$$

(c) For $\mathbf{Z}^{2,1}\in\mathcal{E}_3$, the gradient admits an analysis analogous to that in Lemma H.19. In particular, by Induction H.2 and Lemma H.27, we first obtain the following logit bound:

$$\mathbf{logit}^{(t)}_{5,j}\leq\frac{1}{\mathsf{poly}d}\cdot\Big(1-\mathbf{logit}^{(t)}_{5,j_2}\Big),\qquad j\neq j_2,\tau\big(g_1(y_1)\big).$$

Therefore,

$$\mathbb{E}\Big[\mathcal{N}^{(t)}_{s,2,i}\mathbb{1}_{\mathcal{E}_3}\Big]$$

$$=\mathbb{E}\bigg[(1-\mathbf{logit}^{(t)}_{5,j_2})\cdot\Big(-\mathbf{Attn}^{(t)}_{\mathsf{ans},1\to\mathsf{ans},0}\cdot V_{j_2,r_{g_2\cdot y_1}}(y_1)-\mathbf{Attn}^{(t)}_{\mathsf{ans},1\to\mathsf{pred},1}\cdot V_{j_2,r_{g_2\cdot y_1}}(g_1)$$

$$+\big(1-\mathbf{Attn}^{(t)}_{\mathsf{ans},1\to\mathsf{ans},1}-\mathbf{Attn}^{(t)}_{\mathsf{ans},1\to\mathsf{pred},2}\big)\Lambda^{(t)}_{5,j_2,r_{g_2\cdot y_1}}\pm\widetilde{O}(\sigma_0)\pm\widetilde{O}(\delta^q)\Big)\mathbb{1}_{\tau(x_1)=s}\mathbb{1}_{\mathcal{E}_3}\bigg]$$

$$\geq\Omega\Big(\frac{B}{n_y\cdot d}\Big)\cdot\mathbb{E}\Big[(1-\mathbf{logit}^{(t)}_{5,j_2})\mid\tau(x_1)=s,\mathcal{E}_3\Big].$$

Moving to $\mathcal{N}_{s,2,ii}^{(t)}$, we have

$$\mathbb{E}\Big[\mathcal{N}_{s,2,ii}^{(t)} \mathbb{1}_{\mathcal{E}_3}\Big]$$

$$\geq \mathbb{E}\bigg[\mathbf{logit}_{5,\tau(g_1(y_1))}^{(t)} \cdot \Big(\mathbf{Attn}_{\mathsf{ans},1\to\mathsf{ans},0}^{(t)} \cdot V_{\tau(g_1(y_1)),r_{g_1 \cdot y_1}}(y_1) + \mathbf{Attn}_{\mathsf{ans},1\to\mathsf{pred},1}^{(t)} \cdot V_{\tau(g_1(y_1)),r}(g_1)$$

$$- \big(1 - \mathbf{Attn}_{\mathsf{ans},1\to\mathsf{ans},1}^{(t)} - \mathbf{Attn}_{\mathsf{ans},1\to\mathsf{pred},2}^{(t)}\big)\Lambda_{5,\tau(g_1(y_1)),r_{g_1 \cdot y_1}}^{(t)} \pm \widetilde{O}(\sigma_0) \pm \widetilde{O}(\delta^q)\Big) \mathbb{1}_{\tau(x_1)=s}\mathbb{1}_{\mathcal{E}_3}\bigg]$$

$$- O\Big(\frac{B}{d \cdot \mathsf{poly}d}\Big) \cdot \mathbb{E}\Big[(1 - \mathbf{logit}_{5,j_2}^{(t)}) \mid \tau(x_1) = s,\, \mathcal{E}_3\Big].$$

Combining the two bounds, we conclude

$$\mathbb{E}\Big[\mathcal{N}_{s,2,i}^{(t)} \mathbb{1}_{\mathcal{E}_3}\Big] + \mathbb{E}\Big[\mathcal{N}_{s,2,ii}^{(t)} \mathbb{1}_{\mathcal{E}_3}\Big] \geq \Omega\Big(\frac{B}{n_y \cdot d}\Big) \cdot \mathbb{E}\Big[(1 - \mathbf{logit}_{5,j_2}^{(t)}) \mid \tau(x_1) = s,\, \mathcal{E}_3\Big].$$

(d) For $\mathbf{Z}^{2,1} \in \mathcal{E}_4$, in analogy with Lemma H.19, we obtain the following crude bound:

$$\mathbb{E}\Big[\mathcal{N}_{s,2,i}^{(t)} \mathbb{1}_{\mathcal{E}_4}\Big] + \mathbb{E}\Big[\mathcal{N}_{s,2,ii}^{(t)} \mathbb{1}_{\mathcal{E}_4}\Big] \geq 0.$$

(e) For $\mathbf{Z}^{2,1} \in \mathcal{E}_5 \cup \mathcal{E}_6$, by Induction H.2, Lemma H.29 and Lemma H.30, we obtain the following logit condition:

$$1 - \mathbf{logit}_{5,j_2}^{(t)},\ \ \mathbf{logit}_{5,j}^{(t)} \leq \frac{1}{\mathsf{poly}d} \cdot \mathbb{E}\Big[(1 - \mathbf{logit}_{5,j_2}^{(t)}) \mid \tau(x_1) = s,\, \mathcal{E}_1\Big], \qquad j \neq j_2.$$

Hence, $\mathcal{N}_{s,2,i}^{(t)}$ and $\mathcal{N}_{s,2,ii}^{(t)}$ can be bounded as

$$\Big|\mathbb{E}\big[\mathcal{N}_{s,2,i}^{(t)} \mathbb{1}_{\mathcal{E}_m}\big]\Big|,\ \ \Big|\mathbb{E}\big[\mathcal{N}_{s,2,ii}^{(t)} \mathbb{1}_{\mathcal{E}_m}\big]\Big|$$

$$\leq O\Big(\frac{B}{dn_y} \cdot \frac{1}{\mathsf{poly}d}\Big) \cdot \mathbb{E}\Big[(1 - \mathbf{logit}_{5,j_2}^{(t)}) \mid \tau(x_1) = s,\, \mathcal{E}_1\Big], \qquad m \in \{5, 6\}.$$

Moreover, for $\mathbf{Z}^{2,1} \in \mathcal{E}_6$, if

$$\mathbf{Attn}_{\mathsf{ans},1\to\mathsf{pred},1}^{(t)} + \mathbf{Attn}_{\mathsf{ans},1\to\mathsf{pred},2}^{(t)} > \tfrac{1}{2},$$

then $\mathcal{N}_{s,2,i}^{(t)} = 0$. Nevertheless, due to the above order-wise bound, this observation does not affect the overall analysis.

Putting everything together, we have

$$\Big[-\nabla_{\mathbf{Q}_{4,3}^{(t)}} \mathsf{Loss}_5^{2,2}\Big]_{s,s} + \Big[-\nabla_{\mathbf{Q}_{4,4}^{(t)}} \mathsf{Loss}_5^{2,2}\Big]_{s,s} = \sum_{\kappa \in \{i,ii,iii\}} \sum_{m \in [6]} \mathbb{E}\Big[\mathcal{N}_{s,2,\kappa}^{(t)} \mathbb{1}_{\mathcal{E}_m}\Big]$$

$$\geq \Omega\Big(\frac{B}{d}\Big) \cdot \mathbb{E}\Big[(1 - \mathbf{logit}_{5,j_2}^{(t)}) \mid \tau(x_1) = s,\, \mathcal{E}_1\Big] + \Omega\Big(\frac{B}{n_y d}\Big) \cdot \sum_{m=2}^{3} \mathbb{E}\Big[(1 - \mathbf{logit}_{5,j_2}^{(t)}) \mid \tau(x_1) = s,\, \mathcal{E}_m\Big].$$

$\square$

**Lemma H.32.** *If Induction H.2 holds for all iterations $t \in [T_{1,2,1,s}, T_{1,2,2,s})$, given $s \in \tau(\mathcal{X})$, for $[\mathbf{Q}_{4,p}]_{s,s'}$, $p \in \{3, 4\}$, $s' \neq s \in \tau(\mathcal{X})$, we have*

$$\Big|\Big[-\nabla_{\mathbf{Q}_{4,p}^{(t)}} \mathsf{Loss}_5^{2,2}\Big]_{s,s'}\Big| \leq O\Big(\frac{1}{d}\Big)\Big|\Big[-\nabla_{\mathbf{Q}_{4,p}^{(t)}} \mathsf{Loss}_5^{2,2}\Big]_{s,s}\Big|.$$

### H.4.3 Attention gap is small

**Lemma H.33.** *If Induction H.2 holds for all iterations $t \in [T_{1,2,1,s}, T_{1,2,2,s})$, then for any sample $\mathbf{Z}^{2,1}$, we have*

$$\mathbf{Attn}^{(t)}_{\mathsf{ans},1 \to \mathsf{pred},2} - \mathbf{Attn}^{(t)}_{\mathsf{ans},1 \to \mathsf{ans},1} \leq c_1,$$

*where $c_1 > 0$ is a small constant.*

The proof follows along the same lines as Lemma H.22, and hence the details are omitted for brevity.

**Lemma H.34.** *If Induction H.2 holds for all iterations $t \in [T_{1,2,1,s}, T_{1,2,2,s})$, then for any sample $\mathbf{Z}^{2,1}$, we have*

$$\mathbf{Attn}^{(t)}_{\mathsf{ans},1 \to \mathsf{ans},1} - \mathbf{Attn}^{(t)}_{\mathsf{ans},1 \to \mathsf{pred},2} \leq \mathbf{Attn}^{(t)}_{\mathsf{ans},1 \to \mathsf{pred},1} - c_2,$$

*where $c_2 = \frac{\log d}{4B} > 0$ is a small constant.*

*Proof.* Let $\widetilde{T}$ denote the first time such that

$$\mathbb{E}\Big[\mathbf{Attn}^{(t)}_{\mathsf{ans},1 \to \mathsf{pred},2} + \mathbf{Attn}^{(t)}_{\mathsf{ans},1 \to \mathsf{pred},1} - \mathbf{Attn}^{(t)}_{\mathsf{ans},1 \to \mathsf{ans},1} \mid \tau(x_1) = s\Big] \leq \frac{\log d}{4.02B}.$$

Since $\big|[\mathbf{Q}^{(t)}_{4,p}]_{s,s'}\big| \leq O\left(\frac{[\mathbf{Q}^{(t)}_{4,p}]_{s,s}}{d}\right)$ for $p \in \{3,4\}$, it follows that for any sample $\mathbf{Z}^{2,1}$ with $\tau(x_1) = s$, at time $\widetilde{T}$ we have

$$\mathbf{Attn}^{(t)}_{\mathsf{ans},1 \to \mathsf{pred},2} + \mathbf{Attn}^{(t)}_{\mathsf{ans},1 \to \mathsf{pred},1} - \mathbf{Attn}^{(t)}_{\mathsf{ans},1 \to \mathsf{ans},1} \geq \frac{\log d}{4.02B} \pm \widetilde{O}(1/d).$$

By Lemma H.24, we also have $\mathbf{Attn}^{(\widetilde{T})}_{\mathsf{ans},1 \to \mathsf{pred},2} - \mathbf{Attn}^{(\widetilde{T})}_{\mathsf{ans},1 \to \mathsf{ans},0} \geq \Omega(1) \gg \varrho$. We then consider the following cases:

- If $\mathbf{Z}^{2,1} \in \mathcal{E}_1$, then $\Lambda^{(\widetilde{T})}_{5,j_2,r_{g_2 \cdot y_1}}$ is already activated into the linear regime. Moreover, since $\mathbf{Attn}^{(\widetilde{T})}_{\mathsf{ans},1 \to \mathsf{pred},2} - \mathbf{Attn}^{(\widetilde{T})}_{\mathsf{ans},1 \to \mathsf{ans},1} \leq \frac{\log d}{4.02B} - \mathbf{Attn}^{(t)}_{\mathsf{ans},1 \to \mathsf{pred},1} < 0$, it follows that $\Lambda^{(\widetilde{T})}_{5,j'_2,r_{g_2 \cdot y_0}}$ cannot be activated. Thus, by Lemma H.11,

  $$\mathbb{E}\Big[\big(\mathcal{N}^{(\widetilde{T})}_{s,3,2,i} + \mathcal{N}^{(\widetilde{T})}_{s,3,2,ii} + \mathcal{N}^{(\widetilde{T})}_{s,3,2,iii}\big)\mathbb{1}_{\mathcal{E}_1}\Big] \geq \Omega\big(\tfrac{B}{d}\big) \cdot \mathbb{E}\big[1 - \mathbf{logit}^{(\widetilde{T})}_{5,j_2} \mid \tau(x_1) = s, \mathcal{E}_1\big],$$

  while

  $$\mathbb{E}\Big[\big(\mathcal{N}^{(\widetilde{T})}_{s,4,2,i} + \mathcal{N}^{(\widetilde{T})}_{s,4,2,ii} + \mathcal{N}^{(\widetilde{T})}_{s,4,2,iii}\big)\mathbb{1}_{\mathcal{E}_1}\Big] \leq -\Omega\big(\tfrac{B}{dn_y}\big) \cdot \mathbb{E}\big[1 - \mathbf{logit}^{(\widetilde{T})}_{5,j_2} \mid \tau(x_1) = s, \mathcal{E}_1\big].$$

- If $\mathbf{Z}^{2,1} \in \mathcal{E}_2$, then

  $$\Lambda^{(\widetilde{T})}_{5,j'_2,r_{g_2 \cdot y_0}} = \big(\mathbf{Attn}^{(\widetilde{T})}_{\mathsf{ans},1 \to \mathsf{pred},2} + \mathbf{Attn}^{(\widetilde{T})}_{\mathsf{ans},1 \to \mathsf{pred},1} - \mathbf{Attn}^{(\widetilde{T})}_{\mathsf{ans},1 \to \mathsf{ans},0}\big)2B \leq \frac{\log d}{2.01}.$$

  Hence $\mathbf{logit}^{(1)}_{5,j_2} \leq O(d^{-1.01/2.01})(1 - \mathbf{logit}^{(t)}_{5,j_2})$. By Lemma H.11,

  $$\mathbb{E}\Big[\big(\mathcal{N}^{(\widetilde{T})}_{s,3,2,i} + \mathcal{N}^{(\widetilde{T})}_{s,3,2,ii} + \mathcal{N}^{(\widetilde{T})}_{s,3,2,iii}\big)\mathbb{1}_{\mathcal{E}_2}\Big] \geq \Omega\big(\tfrac{B}{dn_y}\big) \cdot \mathbb{E}\big[1 - \mathbf{logit}^{(\widetilde{T})}_{5,j_2} \mid \tau(x_1) = s, \mathcal{E}_2\big],$$

  and

  $$\mathbb{E}\Big[\big(\mathcal{N}^{(\widetilde{T})}_{s,4,2,i} + \mathcal{N}^{(\widetilde{T})}_{s,4,2,ii} + \mathcal{N}^{(\widetilde{T})}_{s,4,2,iii}\big)\mathbb{1}_{\mathcal{E}_2}\Big] \leq -\Omega\big(\tfrac{B}{dn_y}\big) \cdot \mathbb{E}\big[1 - \mathbf{logit}^{(\widetilde{T})}_{5,j_2} \mid \tau(x_1) = s, \mathcal{E}_2\big].$$

- If $\mathbf{Z}^{2,1} \in \mathcal{E}_3$, we can apply the crude bounds

  $$\mathbb{E}\Big[\big(\mathcal{N}^{(\widetilde{T})}_{s,3,2,i} + \mathcal{N}^{(\widetilde{T})}_{s,3,2,ii} + \mathcal{N}^{(\widetilde{T})}_{s,3,2,iii}\big)\mathbb{1}_{\mathcal{E}_3}\Big] \geq 0,$$

and
$$\mathbb{E}\Big[\big(\mathcal{N}_{s,4,2,i}^{(\widetilde{T})} + \mathcal{N}_{s,4,2,ii}^{(\widetilde{T})} + \mathcal{N}_{s,4,2,iii}^{(\widetilde{T})}\big)\mathbb{1}_{\mathcal{E}_3}\Big] \leq 0.$$

Combining the above, and noting that the gradient contribution from $\mathcal{E}_4 \cup \mathcal{E}_5 \cup \mathcal{E}_6$ is negligible, we conclude

$$\big[-\nabla_{\mathbf{Q}_{4,3}^{(\widetilde{T})}}\mathsf{Loss}_5^{2,2}\big]_{s,s} \geq \Omega\big(\tfrac{B}{d}\big)\cdot\mathbb{E}\big[1 - \mathbf{logit}_{5,j_2}^{(\widetilde{T})} \mid \tau(x_1) = s, \mathcal{E}_1\big] \tag{116}$$
$$+ \Omega\Big(\frac{B}{dn_y}\Big)\cdot\mathbb{E}\big[1 - \mathbf{logit}_{5,j_2}^{(\widetilde{T})} \mid \tau(x_1) = s, \mathcal{E}_2\big],$$

$$\big[-\nabla_{\mathbf{Q}_{4,4}^{(\widetilde{T})}}\mathsf{Loss}_5^{2,2}\big]_{s,s} \leq -\Omega\big(\tfrac{B}{d}\big)\cdot\mathbb{E}\big[1 - \mathbf{logit}_{5,j_2}^{(\widetilde{T})} \mid \tau(x_1) = s, \mathcal{E}_1\big] \tag{117}$$
$$- \Omega\Big(\frac{B}{dn_y}\Big)\cdot\mathbb{E}\big[1 - \mathbf{logit}_{5,j_2}^{(\widetilde{T})} \mid \tau(x_1) = s, \mathcal{E}_2\big].$$

Finally, observe that

$$\mathbf{Attn}_{\mathsf{ans},1\to\mathsf{pred},2}^{(t)} + \mathbf{Attn}_{\mathsf{ans},1\to\mathsf{pred},1}^{(t)} - \mathbf{Attn}_{\mathsf{ans},1\to\mathsf{ans},1}^{(t)} = \frac{e^{[\mathbf{Q}_{4,3}^{(t)}]_{s,s}} - e^{[\mathbf{Q}_{4,4}^{(t)}]_{s,s}} - e^{\widetilde{O}(1/d)}}{e^{[\mathbf{Q}_{4,3}^{(t)}]_{s,s}} + e^{[\mathbf{Q}_{4,4}^{(t)}]_{s,s}} + e^{\widetilde{O}(1/d)}}.$$

Hence, (116) and (117) together imply that

$$\mathbf{Attn}_{\mathsf{ans},1\to\mathsf{pred},2}^{(t)} + \mathbf{Attn}_{\mathsf{ans},1\to\mathsf{pred},1}^{(t)} - \mathbf{Attn}_{\mathsf{ans},1\to\mathsf{ans},1}^{(t)}$$

must decrease at time $\widetilde{T} + 1$. $\qquad\square$

### H.4.4  At the End of Training

**Lemma H.35** (At the end of stage 1.2.2). *Induction H.2 holds for all iterations $T_{1,2,1,s} < t \leq T_{1,2,2,s} = O\Big(\frac{\mathsf{poly}(d)}{\eta B}\Big)$. At the end of training, we have*

(a) *Attention concentration:* $\mathbb{E}[\epsilon_{\mathsf{attn}} \mid \tau(x_1) = s] \leq 2.52c_1$ *for some small constant* $0 < c_1 = \frac{1.005\log d}{2B}$;

(b) *Loss convergence:* $\mathsf{Loss}_{5,s}^{2,2} \leq e^{(-\frac{1}{2}+3.01c_1)\cdot 2B} = \frac{1}{\mathsf{poly}d}$.

*Proof.* Lemma H.24 and Lemma H.31 guarantee the continual growth of $[\mathbf{Q}_{4,3}^{(t)}]$ and $[\mathbf{Q}_{4,4}^{(t)}]$ until the attention weight $\epsilon_{\mathsf{attn}}$ reaches $\frac{4c_1}{3}$. Combining Lemma H.31, Lemma H.32, Lemma H.33, and Lemma H.34, we conclude that there exists a stopping time

$$T_{1,2,2,s} = O\Big(\frac{\mathsf{poly}(d)}{\eta B}\Big),$$

and that Induction H.2 holds for all $t \in [T_{1,2,1,s}, T_{1,2,2,s})$.

Next, we establish an upper bound for $\epsilon_{\mathsf{attn}}$ at the end of training. In this stage, it suffices to focus on the main event $\mathcal{E}_1$. We denote

$$\mathbf{Attn}_{\mathsf{ans},1\to\mathsf{pred},1}^{(t)}, \quad \mathbf{Attn}_{\mathsf{ans},1\to\mathsf{ans},1}^{(t)} = c + \widetilde{O}\big(\tfrac{1}{d}\big).$$

- If $\mathbf{Attn}_{\mathsf{ans},1\to\mathsf{pred},2}^{(t)} \geq \mathbf{Attn}_{\mathsf{ans},1\to\mathsf{ans},1}^{(t)}$, then for $\mathbf{Z}^{2,1} \in \mathcal{E}_1$ we have

$$\log p_F(\mathbf{Z}_{\mathsf{ans},2,5}|\mathbf{Z}^{(2,1)}) \geq \Theta(1)\cdot e^{\log d - \big(\mathbf{Attn}_{\mathsf{ans},1\to\mathsf{pred},2}^{(t)} - \mathbf{Attn}_{\mathsf{ans},1\to\mathsf{ans},0}^{(t)}\big)2B}$$
$$\geq \Theta\left(e^{-\big(\frac{1}{2}+\frac{c_1}{2}-2c-\frac{\log d}{B}\big)2B}\right),$$

where the last inequality follows from

$$\mathbf{Attn}_{\mathsf{ans},1\to\mathsf{pred},2}^{(t)} \leq \tfrac{1}{2}(1 - 2c + c_1) = \tfrac{1+c_1}{2} - c.$$

- If $\mathbf{Attn}^{(t)}_{\mathsf{ans},1\to\mathsf{pred},2} \leq \mathbf{Attn}^{(t)}_{\mathsf{ans},1\to\mathsf{ans},1}$, then for $\mathbf{Z}^{2,1} \in \mathcal{E}_1$ we have

$$\log p_F(\mathbf{Z}_{\mathsf{ans},2,5}|\mathbf{Z}^{(2,1)}) \geq \Theta(1) \cdot e^{\log d - \left(\mathbf{Attn}^{(t)}_{\mathsf{ans},1\to\mathsf{pred},2} - \mathbf{Attn}^{(t)}_{\mathsf{ans},1\to\mathsf{ans},0}\right)2B}$$
$$\geq \Theta\left(e^{-\left(\frac{1}{2} - 2c - \frac{\log d}{B}\right)2B}\right),$$

where the last inequality follows from the fact that $\mathbf{Attn}^{(t)}_{\mathsf{ans},1\to\mathsf{pred},2} \leq \frac{1}{2} - c$.

Putting the above cases together, we obtain

$$\tfrac{1}{2} + \tfrac{c_1}{2} - 2c - \tfrac{\log d}{B} \ \geq \ \tfrac{1}{2} - 3.01c_1,$$

which implies

$$c \ \leq \ \tfrac{1}{2}(3.02c_1 + 0.5c_1 - c_1) \ = \ 1.26c_1.$$

$\square$

# I  Recursive Learning the Attention Layer: Symmetry Case

In this section, we analyze the recursive learning dynamics of the attention layer in the symmetry case. To this end, we recall the definitions of the greedy data annotator, the bootstrapped LEGO data distribution, and the associated self-training loss.

**Definition I.1** (Greedy data annotator). The greedy language model $\widehat{p}_F$ induced by a network $F$ is defined as

$$\widehat{p}_F(Z_{\mathsf{ans},L'+1} \mid Z^{L,L'}) = \begin{cases} 1, & \text{if } Z_{\mathsf{ans},L'+1} = \mathrm{argmax}_Z \, p_F(Z \mid Z^{L,L'}), \\ 0, & \text{otherwise.} \end{cases} \tag{118}$$

**Definition I.2** (Bootstrapped LEGO distribution). We define $\mathcal{D}_F^{L,L'}$ as the LEGO distribution in Assumption 3.2, except that the answers $Z_{\mathsf{ans},\ell}, 1 \leq \ell \leq L'$ are generated recursively by sampling $Z_{\mathsf{ans},\ell} \sim \widehat{p}_F(\cdot \mid Z^{L,\ell-1})$ from the greedy language model $\widehat{p}_F$.

**Definition I.3** (Self-training loss). Given a (fixed) model $\widetilde{F}$ and length $L$, the self-training next-clause-prediction loss is defined by replacing $\mathcal{D}^{L,L'}$ in Definition 3.7 with the bootstrapped distribution $\mathcal{D}_F^{L,L'}$ from Definition I.2:

$$\mathsf{Loss}^{L,L'}_{\widetilde{F}}(F) \triangleq \mathbb{E}_{Z^{L,L'}\sim\mathcal{D}^{L,L'}_{\widetilde{F}}}\left[ - \log p_F(Z_{\mathsf{ans},L',i} \mid Z^{L,L'-1})\right]. \tag{119}$$

Similarly, the per-token loss is defined by

$$\mathsf{Loss}^{L,L'}_{\widetilde{F},i} = \mathbb{E}_{Z^{L,L'}\sim\mathcal{D}^{L,L'}_{\widetilde{F}}}\left[ - \log p_{F_i}(Z_{\mathsf{ans},L',i} \mid Z^{L,L'-1})\right], \quad i \in [5].$$

At stage $k \geq 2$, let $F^{(T_{k-1})}$ denote the model obtained from the previous stage, trained on the task $\mathcal{T}^{L_{k-1}}$ with $L_{k-1} = 2^{k-1}$. Using the greedy annotator $\widehat{p}_{F^{(T_{k-1})}}$, we construct the bootstrapped LEGO distribution $\mathcal{D}^{L_k,L_k}_{F^{(T_{k-1})}}$ with total length $L_k = 2^k$. The corresponding self-training loss $\mathsf{Loss}^{L_k,L'}_{F^{(T_{k-1})}}$ is then defined as in Definition I.3. In this stage, we focus on training via gradient descent on $\mathsf{Loss}^{L_k,L'}_{F^{(T_{k-1})},i}$ with sequence length $L' = 2$ and target token $i = 5$, initialized from the previous-stage model $F^{(T_{k-1})}$.

For notational simplicity, throughout the discussion we drop the subscript of the expectation operator $\mathbb{E}$ when $\mathbf{Z}$ is sampled from the ground-truth LEGO distribution; the subscript will be explicitly included only when the expectation is taken over the bootstrapped distribution. Moreover, we abbreviate the wrong attention error $\epsilon^{L_k,2}_{\mathsf{attn}}$ as $\epsilon^{L_k}_{\mathsf{attn}}$ and the attention gap $\Delta^{L_k,2}$ as $\Delta^{L_k}$ when the context is clear.

## I.1 Preliminaries

We first present preliminaries on the recursive learning attention layer, including its gradient computations and some useful probability lemmas.

### I.1.1 Gradient Computations

**Fact I.1** (Gradients of $\mathbf{Q}$). *Given $F^{(T_{k-1})}$ from the previous stage $\mathcal{T}^{L_{k-1}}$ with $L_{k-1} = 2^{k-1}$ for $k \geq 2$, for $(p, q) \in \left\{ (4, 3), (4, 4) \right\}$, we have*

$$-\nabla_{\mathbf{Q}_{p,q}} \mathsf{Loss}^{L_k,2}_{F^{(T_{k-1})},5} = -\nabla_{\mathbf{Q}_{p,q}} \mathsf{Loss}^{L_k,2}_5.$$

**Remark I.1.** *This fact is straightforward to verify. Since the attention error $\epsilon^{L_{k-1},2}_{\mathsf{attn}}$ has already been controlled at a small constant level in the previous stage for the model $F^{(T_{k-1})}$, when transitioning from length $2^{k-1}$ to $2^k$, the incorrect attention produced by $F^{(T_{k-1})}$ cannot increase significantly. In particular, we have*

$$\epsilon^{L_k}_{\mathsf{attn}} \leq 2\epsilon^{L_{k-1}}_{\mathsf{attn}},$$

*which remains small. Moreover, the attention gap $\Delta^{L_k} = |\mathbf{Attn}_{\mathsf{ans},1\to\mathsf{pred},2} - \mathbf{Attn}_{\mathsf{ans},1\to\mathsf{ans},1}|$ is non-increasing due to the doubling of irrelevant clauses. As a result, the probability assigned by $F^{(T_{k-1})}$ to the correct prediction of $Z_{\mathsf{ans},2,5}$ given $Z^{L_k,1}$ remains significantly larger than that of any incorrect prediction. Consequently, under the greedy data annotator, the self-training loss $\mathsf{Loss}^{L_k,2}_{F^{(T_{k-1})},5}$ coincides with the original loss $\mathsf{Loss}^{L_k,2}_5$ on the ground-truth LEGO distribution.*

**Lemma I.1** (Gradients of $\mathbf{Q}_{4,3}$). *Given $F^{(T_{k-1})}$ from the previous stage and $s \in \tau(\mathcal{X})$, for the diagonal entry $[\mathbf{Q}_{4,3}]_{s,s}$ of the block $\mathbf{Q}_{4,3}$, we have*

$$\left[ -\nabla_{\mathbf{Q}_{4,3}} \mathsf{Loss}^{L_k,2}_{F^{(T_{k-1})},5} \right]_{s,s} = \mathbb{E}\Bigg[ \mathbf{Attn}_{\mathsf{ans},1\to\mathsf{pred},2} \cdot \Bigg( \sum_{j\in[d]} \mathcal{E}_{5,j}(\mathbf{Z}^{2,1}) \sum_{r\in[m]} \mathbf{sReLU}'(\Lambda_{5,j,r})\cdot$$
$$\left( \langle \mathbf{W}_{5,j,r}, \mathbf{Z}_{\mathsf{pred},2} \rangle - \Lambda_{5,j,r} + b_{5,j,r} \right) \Bigg) \mathbb{1}_{s=\tau(x_1)} \Bigg].$$

*Moreover, for the off-diagonal entries $[\mathbf{Q}_{4,3}]_{s,s'}$ with $s \neq s'$, we have*

$$\left[ -\nabla_{\mathbf{Q}_{4,3}} \mathsf{Loss}^{L_k,2}_{F^{(T_{k-1})},5} \right]_{s,s'} = \mathbb{E}\Bigg[ \mathbf{Attn}_{\mathsf{ans},1\to\mathsf{pred},1} \cdot \Bigg( \sum_{j\in[d]} \mathcal{E}_{5,j}(\mathbf{Z}^{2,1}) \sum_{r\in[m]} \mathbf{sReLU}'(\Lambda_{5,j,r})\cdot$$
$$\left( \langle \mathbf{W}_{5,j,r}, \mathbf{Z}_{\mathsf{pred},1} \rangle - \Lambda_{5,j,r} + b_{5,j,r} \right) \Bigg) \mathbb{1}_{s=\tau(x_1),s'=\tau(x_0)} \Bigg].$$

**Lemma I.2** (Gradients of $\mathbf{Q}_{4,4}$). *Given $F^{(T_{k-1})}$ from the previous stage and $s \in \tau(\mathcal{X})$, for the diagonal entry $[\mathbf{Q}_{4,4}]_{s,s}$ of the block $\mathbf{Q}_{4,4}$, we have*

$$\left[ -\nabla_{\mathbf{Q}_{4,4}} \mathsf{Loss}^{L_k,2}_{F^{(T_{k-1})},5} \right]_{s,s} = \mathbb{E}\Bigg[ \mathbf{Attn}_{\mathsf{ans},1\to\mathsf{ans},1} \cdot \Bigg( \sum_{j\in[d]} \mathcal{E}_{5,j}(\mathbf{Z}^{2,1}) \sum_{r\in[m]} \mathbf{sReLU}'(\Lambda_{5,j,r})\cdot$$
$$\left( \langle \mathbf{W}_{5,j,r}, \mathbf{Z}_{\mathsf{ans},1} \rangle - \Lambda_{5,j,r} + b_{5,j,r} \right) \Bigg) \mathbb{1}_{s=\tau(x_1)} \Bigg].$$

*Moreover, for the off-diagonal entries $[\mathbf{Q}_{4,4}]_{s,s'}$ with $s \neq s'$, we have*

$$\left[ -\nabla_{\mathbf{Q}_{4,4}} \mathsf{Loss}^{L_k,2}_{F^{(T_{k-1})},5} \right]_{s,s'} = \mathbb{E}\Bigg[ \mathbf{Attn}_{\mathsf{ans},1\to\mathsf{ans},0} \cdot \Bigg( \sum_{j\in[d]} \mathcal{E}_{5,j}(\mathbf{Z}^{2,1}) \sum_{r\in[m]} \mathbf{sReLU}'(\Lambda_{5,j,r})\cdot$$
$$\left( \langle \mathbf{W}_{5,j,r}, \mathbf{Z}_{\mathsf{ans},0} \rangle - \Lambda_{5,j,r} + b_{5,j,r} \right) \Bigg) \mathbb{1}_{s=\tau(x_1),s'=\tau(x_0)} \Bigg].$$

**Notations for activated neurons.** Recall that

$$\mathfrak{A} = \bigcup_{j \in \tau(\mathcal{Y})} \mathfrak{A}_j, \quad \text{where } \mathfrak{A}_j = \{\, r_{j,y} \mid y \in \mathcal{Y} \,\}.$$

Given $\mathbf{Z}^{L,\ell-1}$, let

$$\widehat{\mathcal{G}}(\mathbf{Z}^{L,\ell-1}) = \bigcup_{\ell'=1}^{L} \{g_{\ell'}\}$$

be the collection of all group elements chosen in the predicate clauses, and similarly define

$$\widehat{\mathcal{Y}} = \bigcup_{\ell'=0}^{\ell-1} \{y_{\ell'}\}.$$

We then introduce

$$\widehat{\mathfrak{A}}_j(\mathbf{Z}^{L,\ell-1}) = \left\{\, r_{g\cdot y} \,\Big|\, g \in \text{Fiber}_{j,y}, \; g \in \widehat{\mathcal{G}}(\mathbf{Z}^{L,\ell-1}) \vee y \in \widehat{\mathcal{Y}} \right\}.$$

For simplicity, we omit the dependence on $\mathbf{Z}^{L,\ell-1}$ in the notation of $\widehat{\mathfrak{A}}_j$ when it is clear from context. The above recalls the relevant notations from Appendix H.2. With these preliminaries in place, we now state the following lemma, which also relies on the feature-magnitude bounds established therein.

**Lemma I.3** (Characterizations of Lambda). *Given $\mathbf{Z}^{L_k,1}$ with $\{\mathbf{Attn}_{\mathsf{ans},1\to k}\}_{\mathbf{k}\in\mathcal{I}^{L_k,1}}$, then*

*(a) for $j \in \tau(\mathcal{Y})$, for activated neuron $r \in \mathfrak{A}_j$, we have*

$$\Lambda_{5,j,r} = \sum_{\ell'=1}^{2} \mathbf{Attn}_{\mathsf{ans},1\to\mathsf{pred},\ell'} V_{j,r}(g_{\ell'}) + \sum_{\ell'=1}^{L_k} \mathbf{Attn}_{\mathsf{ans},1\to\mathsf{ans},\ell'-1} V_{j,r}(y_{\ell'-1}) \pm \widetilde{O}(\sigma_0).$$

*(b) for $j \in \tau(\mathcal{Y})$, for any non-activated neuron $r \notin \mathfrak{A}_j$ we have*

$$\left|\Lambda_{5,j,r}\right| \leq O(\delta).$$

*(c) for $j \notin \tau(\mathcal{Y})$, for any $r \in [m]$, we have*

$$\left|\Lambda_{5,j,r}\right| \leq \widetilde{O}(\sigma_0).$$

**Lemma I.4.** *Given $j \in \tau(\mathcal{Y})$, for $r \in \mathfrak{A}_j \setminus \widehat{\mathfrak{A}}_j$, we have $\mathbf{sReLU}'(\Lambda_{5,j,r}) = 0$.*

We are now ready to derive the gradients of the attention layer, building on Lemmas I.1 and I.2 and the properties established above.

**Lemma I.5** (Refined expression for the gradient of $\mathbf{Q}_{4,3}$). *Given $F^{(T_k-1)}$ from the previous stage and $s \in \tau(\mathcal{X})$, for the diagonal entry $[\mathbf{Q}_{4,3}]_{s,s}$ of the block $\mathbf{Q}_{4,3}$, letting $j_2 = \tau(g_2(y_1))$, we have*

$$\left[-\nabla_{\mathbf{Q}_{4,3}}\mathsf{Loss}_{F^{(T_k-1)},5}^{L_k,2}\right]_{s,s} = \mathbb{E}\Bigg[\mathbf{Attn}_{\mathsf{ans},1\to\mathsf{pred},2}\cdot$$

$$\left((1-\mathbf{logit}_{5,j_2})\cdot\left(\sum_{r\in\widehat{\mathfrak{A}}_{j_2}}\mathbf{sReLU}'(\Lambda_{5,j_2,r})\cdot\left(V_{j_2,r}(g_2)-\Lambda_{5,j_2,r}\pm\widetilde{O}(\sigma_0)\right)\pm\widetilde{O}(\delta^q)\right)\right.$$

$$-\sum_{j\neq j_2\in\tau(\mathcal{Y})}\mathbf{logit}_{5,j}\cdot\left(\sum_{r\in\widehat{\mathfrak{A}}_j}\mathbf{sReLU}'(\Lambda_{5,j,r})\cdot\left(V_{j,r}(g_2)-\Lambda_{5,j,r}\pm\widetilde{O}(\sigma_0)\right)\pm\widetilde{O}(\delta^q)\right)$$

$$\left.\pm\sum_{j\notin\tau(\mathcal{Y})}\mathbf{logit}_{5,j}\widetilde{O}(\sigma_0^q)\right)\mathbb{1}_{\tau(x_1)=s}\Bigg].$$

*Moreover, for the off-diagonal entries $[\mathbf{Q}_{4,3}]_{s,s'}$ with $s \neq s'$, we have*

$$\left[ -\nabla_{\mathbf{Q}_{4,3}} \mathsf{Loss}^{L_k,2}_{F^{(T_{k-1})},5} \right]_{s,s'} = \mathbb{E}\left[ \mathbf{Attn}_{\mathsf{ans},1\to\mathsf{pred},1} \cdot \right.$$

$$\left( (1 - \mathbf{logit}_{5,j_2}) \cdot \left( \sum_{r \in \widehat{\mathfrak{A}}_{j_2}} \mathbf{sReLU}'(\Lambda_{5,j_2,r}) \cdot \left( V_{j_2,r}(g_1) - \Lambda_{5,j_2,r} \pm \widetilde{O}(\sigma_0) \right) \pm \widetilde{O}(\delta^q) \right) \right.$$

$$- \sum_{j \neq j_2 \in \tau(\mathcal{Y})} \mathbf{logit}_{5,j} \cdot \left( \sum_{r \in \widehat{\mathfrak{A}}_j} \mathbf{sReLU}'(\Lambda_{5,j,r}) \cdot \left( V_{j,r}(g_1) - \Lambda_{5,j,r} \pm \widetilde{O}(\sigma_0) \right) \pm \widetilde{O}(\delta^q) \right)$$

$$\left. \left. \pm \sum_{j \notin \tau(\mathcal{Y})} \mathbf{logit}_{5,j} \widetilde{O}(\sigma_0^q) \right) \mathbb{1}_{\tau(x_1)=s, \tau(x_0)=s'} \right].$$

**Lemma I.6** (Refined expression for the gradient of $\mathbf{Q}_{4,4}$). *Given $F^{(T_{k-1})}$ from the previous stage and $s \in \tau(\mathcal{X})$, for the diagonal entry $[\mathbf{Q}_{4,4}]_{s,s}$ of the block $\mathbf{Q}_{4,4}$, letting $j_2 = \tau(g_2(y_1))$, we have*

$$\left[ -\nabla_{\mathbf{Q}_{4,4}} \mathsf{Loss}^{L_k,2}_{F^{(T_{k-1})},5} \right]_{s,s} = \mathbb{E}\left[ \mathbf{Attn}_{\mathsf{ans},1\to\mathsf{ans},1} \cdot \right.$$

$$\left( (1 - \mathbf{logit}_{5,j_2}) \cdot \left( \sum_{r \in \widehat{\mathfrak{A}}_{j_2}} \mathbf{sReLU}'(\Lambda_{5,j_2,r}) \cdot \left( V_{j_2,r}(y_1) - \Lambda_{5,j_2,r} \pm \widetilde{O}(\sigma_0) \right) \pm \widetilde{O}(\delta^q) \right) \right.$$

$$- \sum_{j \neq j_2 \in \tau(\mathcal{Y})} \mathbf{logit}_{5,j} \cdot \left( \sum_{r \in \widehat{\mathfrak{A}}_j} \mathbf{sReLU}'(\Lambda_{5,j,r}) \cdot \left( V_{j,r}(y_1) - \Lambda_{5,j,r} \pm \widetilde{O}(\sigma_0) \right) \pm \widetilde{O}(\delta^q) \right)$$

$$\left. \left. \pm \sum_{j \notin \tau(\mathcal{Y})} \mathbf{logit}_{5,j} \widetilde{O}(\sigma_0^q) \right) \mathbb{1}_{\tau(x_1)=s} \right].$$

*Moreover, for the off-diagonal entries $[\mathbf{Q}_{4,3}]_{s,s'}$ with $s \neq s'$, we have*

$$\left[ -\nabla_{\mathbf{Q}_{4,4}} \mathsf{Loss}^{L_k,2}_{F^{(T_{k-1})},5} \right]_{s,s'} = \mathbb{E}\left[ \mathbf{Attn}_{\mathsf{ans},1\to\mathsf{ans},0} \cdot \right.$$

$$\left( (1 - \mathbf{logit}_{5,j_2}) \cdot \left( \sum_{r \in \widehat{\mathfrak{A}}_{j_2}} \mathbf{sReLU}'(\Lambda_{5,j_2,r}) \cdot \left( V_{j_2,r}(y_0) - \Lambda_{5,j_2,r} \pm \widetilde{O}(\sigma_0) \right) \pm \widetilde{O}(\delta^q) \right) \right.$$

$$- \sum_{j \neq j_2 \in \tau(\mathcal{Y})} \mathbf{logit}_{5,j} \cdot \left( \sum_{r \in \widehat{\mathfrak{A}}_j} \mathbf{sReLU}'(\Lambda_{5,j,r}) \cdot \left( V_{j,r}(y_0) - \Lambda_{5,j,r} \pm \widetilde{O}(\sigma_0) \right) \pm \widetilde{O}(\delta^q) \right)$$

$$\left. \left. \pm \sum_{j \notin \tau(\mathcal{Y})} \mathbf{logit}_{5,j} \widetilde{O}(\sigma_0^q) \right) \mathbb{1}_{\tau(x_1)=s, \tau(x_0)=s'} \right].$$

Following the above calculations, we can further obtain the gradient summation of $\mathbf{Q}_{4,3}$ and $\mathbf{Q}_{4,4}$ as follows:

**Lemma I.7** (Gradient sum of $\mathbf{Q}_{4,3}$ and $\mathbf{Q}_{4,4}$). *Given $F^{(T_{k-1})}$ from the previous stage and $s \in \tau(\mathcal{X})$, letting $j_2 = \tau(g_2(y_1))$, we have*

$$\left[ -\nabla_{\mathbf{Q}_{4,3}} \mathsf{Loss}^{L_k,2}_{F^{(T_{k-1})},5} \right]_{s,s} + \left[ -\nabla_{\mathbf{Q}_{4,3}} \mathsf{Loss}^{L_k,2}_{F^{(T_{k-1})},5} \right]_{s,s}$$

$$= \mathbb{E}\left[ \left( (1 - \mathbf{logit}_{5,j_2}) \cdot \left( \sum_{r \in \widehat{\mathfrak{A}}_{j_2}} \mathbf{sReLU}'(\Lambda_{5,j_2,r}) \cdot \right. \right. \right.$$

$$\left( -\mathbf{Attn}_{\mathsf{ans},1\to\mathsf{ans},0} \cdot V_{j_2,r}(y_0) - \sum_{\ell \neq 2} \mathbf{Attn}_{\mathsf{ans},1\to\mathsf{pred},\ell} \cdot V_{j_2,r}(g_\ell) \right)$$

$$+ \left(1 - \mathbf{Attn}_{\mathsf{ans},1\to\mathsf{ans},1} - \mathbf{Attn}_{\mathsf{ans},1\to\mathsf{pred},2}\right)\Lambda_{5,j_2,r} \pm \widetilde{O}(\sigma_0)\right) \pm \widetilde{O}(\delta^q)\right)$$

$$+ \sum_{j\neq j_2\in\tau(\mathcal{Y})} \mathbf{logit}_{5,j} \cdot \left( \sum_{r\in\widehat{\mathfrak{A}}_j} \mathbf{sReLU}'(\Lambda_{5,j,r})\cdot\right.$$

$$\left(\mathbf{Attn}_{\mathsf{ans},1\to\mathsf{ans},0} \cdot V_{j,r}(y_0) + \sum_{\ell\neq 2}\mathbf{Attn}_{\mathsf{ans},1\to\mathsf{pred},\ell} \cdot V_{j_2,r}(g_\ell)\right.$$

$$\left.- \left(1 - \mathbf{Attn}_{\mathsf{ans},1\to\mathsf{ans},1} - \mathbf{Attn}_{\mathsf{ans},1\to\mathsf{pred},2}\right)\Lambda_{5,j,r} \pm \widetilde{O}(\sigma_0)\right) \pm \widetilde{O}(\delta^q)\right)$$

$$\left.\pm \sum_{j\notin\tau(\mathcal{Y})}\mathbf{logit}_{5,j} \cdot \left(\mathbf{Attn}_{\mathsf{ans},1\to\mathsf{ans},1} + \mathbf{Attn}_{\mathsf{ans},1\to\mathsf{pred},2}\right)\widetilde{O}(\sigma_0^q)\right)\mathbb{1}_{\tau(x_1)=s}\right].$$

**Notations for gradient decompositions.** Next we define some useful notations to further simplify the expressions of gradient.

**Lemma I.8.** *Given $F^{(T_{k-1})}$ from the previous stage and $s \in \tau(\mathcal{X})$, we define the following notations for the gradient decompositions:*

*1. for $[\mathbf{Q}_{4,3}]_{s,s}$ we have $\left[-\nabla_{\mathbf{Q}_{4,3}}\mathsf{Loss}^{L_k,2}_{F^{(T_{k-1})},5}\right]_{s,s} = \mathbb{E}\left[\mathcal{N}_{s,3,L_k,i} + \mathcal{N}_{s,3,L_k,ii} + \mathcal{N}_{s,3,L_k,iii}\right]$, where*

$$\mathcal{N}_{s,3,L_k,i} = \mathbf{Attn}_{\mathsf{ans},1\to\mathsf{pred},2} \cdot (1 - \mathbf{logit}_{5,j_2})\cdot \tag{120}$$
$$\left( \sum_{r\in\widehat{\mathfrak{A}}_{j_2}} \mathbf{sReLU}'(\Lambda_{5,j_2,r}) \cdot \left(V_{j_2,r}(g_2) - \Lambda_{5,j_2,r} \pm \widetilde{O}(\sigma_0)\right) \pm \widetilde{O}(\delta^q)\right)\mathbb{1}_{\tau(x_1)=s},$$

$$\mathcal{N}_{s,3,L_k,ii} = -\mathbf{Attn}_{\mathsf{ans},1\to\mathsf{pred},2} \cdot \sum_{j\neq j_2\in\tau(\mathcal{Y})} \mathbf{logit}_{5,j}\cdot \tag{121}$$
$$\left( \sum_{r\in\widehat{\mathfrak{A}}_j} \mathbf{sReLU}'(\Lambda_{5,j,r}) \cdot \left(V_{j,r}(g_2) - \Lambda_{5,j,r} \pm \widetilde{O}(\sigma_0)\right) \pm \widetilde{O}(\delta^q)\right)\mathbb{1}_{\tau(x_1)=s},$$

$$\mathcal{N}_{s,3,L_k,iii} = \pm\mathbf{Attn}_{\mathsf{ans},1\to\mathsf{pred},2} \cdot \sum_{j\notin\tau(\mathcal{Y})} \mathbf{logit}_{5,j}\widetilde{O}(\sigma_0^q)\mathbb{1}_{\tau(x_1)=s}.$$
$$\tag{122}$$

*2. for $[\mathbf{Q}_{4,4}]_{s,s}$, we have $\left[-\nabla_{\mathbf{Q}_{4,4}}\mathsf{Loss}^{L_k,2}_{F^{(T_{k-1})},5}\right]_{s,s} = \mathbb{E}\left[\mathcal{N}_{s,4,L_k,i} + \mathcal{N}_{s,4,L_k,ii} + \mathcal{N}_{s,4,L_k,iii}\right]$, where*

$$\mathcal{N}_{s,4,L_k,i} = \mathbf{Attn}_{\mathsf{ans},1\to\mathsf{ans},1} \cdot (1 - \mathbf{logit}_{5,j_2})\cdot \tag{123}$$
$$\left( \sum_{r\in\widehat{\mathfrak{A}}_{j_2}} \mathbf{sReLU}'(\Lambda_{5,j_2,r}) \cdot \left(V_{j_2,r}(y_1) - \Lambda_{5,j_2,r} \pm \widetilde{O}(\sigma_0)\right) \pm \widetilde{O}(\delta^q)\right)\mathbb{1}_{\tau(x_1)=s},$$

$$\mathcal{N}_{s,4,L_k,ii} = -\mathbf{Attn}_{\mathsf{ans},1\to\mathsf{ans},1} \cdot \sum_{j\neq j_2\in\tau(\mathcal{Y})} \mathbf{logit}_{5,j}\cdot \tag{124}$$
$$\left( \sum_{r\in\widehat{\mathfrak{A}}_j} \mathbf{sReLU}'(\Lambda_{5,j,r}) \cdot \left(V_{j,r}(y_1) - \Lambda_{5,j,r} \pm \widetilde{O}(\sigma_0)\right) \pm \widetilde{O}(\delta^q)\right)\mathbb{1}_{\tau(x_1)=s},$$

$$\mathcal{N}_{s,4,L_k,iii} = \pm\mathbf{Attn}_{\mathsf{ans},1\to\mathsf{ans},1} \cdot \sum_{j\notin\tau(\mathcal{Y})} \mathbf{logit}_{5,j}\widetilde{O}(\sigma_0^q)\mathbb{1}_{\tau(x_1)=s}.$$
$$\tag{125}$$

3. *for the summation of* $[\mathbf{Q}_{4,3}]_{s,s}$ *and* $[\mathbf{Q}_{4,4}]_{s,s}$, *we have* $\left[ - \nabla_{\mathbf{Q}_{4,3}}\mathsf{Loss}_5^{2,2}\right]_{s,s} + \left[ - \nabla_{\mathbf{Q}_{4,4}}\mathsf{Loss}_5^{2,2}\right]_{s,s} = \mathbb{E}\left[\mathcal{N}_{s,2,i} + \mathcal{N}_{s,2,ii} + \mathcal{N}_{s,2,iii}\right]$, *where*

$$\mathcal{N}_{s,L_k,i} = (1 - \mathbf{logit}_{5,j_2}) \cdot \Big( \sum_{r \in \widehat{\mathfrak{A}}_{j_2}} \mathbf{sReLU}'(\Lambda_{5,j_2,r}) \cdot$$

$$\Big( - \mathbf{Attn}_{\mathsf{ans},1\to\mathsf{ans},0} \cdot V_{j_2,r}(y_0) - \sum_{\ell\neq 2} \mathbf{Attn}_{\mathsf{ans},1\to\mathsf{pred},\ell} \cdot V_{j_2,r}(g_\ell)$$

$$+ \big(1 - \mathbf{Attn}_{\mathsf{ans},1\to\mathsf{ans},1} - \mathbf{Attn}_{\mathsf{ans},1\to\mathsf{pred},2}\big)\Lambda_{5,j_2,r} \pm \widetilde{O}(\sigma_0)\Big) \pm \widetilde{O}(\delta^q)\Big)\mathbb{1}_{\tau(x_1)=s},$$

$$\mathcal{N}_{s,L_k,ii} = \sum_{j\neq j_2 \in \tau(\mathcal{Y})} \mathbf{logit}_{5,j} \cdot \Big( \sum_{r \in \widehat{\mathfrak{A}}_j} \mathbf{sReLU}'(\Lambda_{5,j,r}) \cdot$$

$$\Big( \mathbf{Attn}_{\mathsf{ans},1\to\mathsf{ans},0} \cdot V_{j,r}(y_0) + \sum_{\ell\neq 2} \mathbf{Attn}_{\mathsf{ans},1\to\mathsf{pred},\ell} \cdot V_{j,r}(g_\ell)$$

$$- \big(1 - \mathbf{Attn}_{\mathsf{ans},1\to\mathsf{ans},1} - \mathbf{Attn}_{\mathsf{ans},1\to\mathsf{pred},2}\big)\Lambda_{5,j,r} \pm \widetilde{O}(\sigma_0)\Big) \pm \widetilde{O}(\delta^q)\Big)\mathbb{1}_{\tau(x_1)=s},$$

$$\mathcal{N}_{s,L_k,iii} = \pm \sum_{j\notin\tau(\mathcal{Y})} \mathbf{logit}_{5,j} \cdot \big(\mathbf{Attn}_{\mathsf{ans},1\to\mathsf{ans},1} + \mathbf{Attn}_{\mathsf{ans},1\to\mathsf{pred},2}\big)\widetilde{O}(\sigma_0^q)\mathbb{1}_{\tau(x_1)=s}.$$

### I.1.2 Probabilistic Events

We introduce the following events:

$$\mathcal{E}_{L_k,1} = \Big\{ \frac{1}{L_k} \sum_{\ell\in[L_k]\backslash\{2\}} \mathbb{1}\{g_\ell(y_1) = g_2(y_1)\} \leq U_k \Big\},$$

$$\mathcal{E}_{L_k,2} = \big\{y_0 = y_1\big\},$$

$$\mathcal{E}_{L_k,3} = \Big\{ \max_{y\in\mathcal{Y}} \sum_{\ell\in[L_k]\backslash\{2\}} \mathbb{1}\{g_\ell(y) = g_2(y)\} \leq W_k \Big\}.$$

We set

$$U_k = \frac{1}{L_k}\Big\lceil 1 + \frac{L_k}{8}\Big\rceil = \Theta(1), \qquad W_k = \begin{cases} \Theta\Big(\dfrac{\log n_y}{\log\big(\frac{4n_y}{L_k}\log n_y\big)}\Big), & L_k \leq n_y \log n_y, \\ \Theta\big(\dfrac{L_k}{n_y}\big), & L_k \geq n_y \log n_y. \end{cases}$$

By standard balls-into-bins tail bounds and maximum-load estimates (e.g., [109]), we obtain

$$\mathbf{Pr}\big(\mathbf{Z}^{L_k,1} \notin \mathcal{E}_{L_k,1}\big) = \begin{cases} O\big(n_y^{-U_k L_k + 1}\big), & L_k \ll n_y, \\ \exp\big(-\Theta(n_y \log n_y)\big), & L_k = \Theta(n_y), \\ \exp\big(-\Theta(L_k \log n_y)\big), & L_k \gg n_y, \end{cases}$$

and

$$\mathbf{Pr}\big(\mathbf{Z}^{L_k,1} \in \mathcal{E}_{L_k,3}\big) = 1 - n_y^{-\Omega(1)}.$$

### I.2 Reducing the Wrong Attention

As discussed in Remark I.1, both $\epsilon_{\mathsf{attn}}^{L_k}$ and the attention gap remain bounded by a small constant at the beginning of stage $k$. Hence, we can directly proceed to stage 1.2.2 of the convergence analysis as $\mathcal{T}^2$. Moreover, by the symmetry between $[\mathbf{Q}_{4,3}]_{s,s}$ and $[\mathbf{Q}_{4,4}]_{s,s}$, we may, without loss of generality, restrict attention to a particular $s \in \tau(\mathcal{X})$ and analyze the corresponding loss $\mathsf{Loss}_{5,s}^{L_k,2}$ in what follows.

**Induction I.1.** *Given* $F^{(T_{k-1})}$ *from the previous stage* $\mathcal{T}^{L_{k-1}}$ *with* $L_{k-1} = 2^{k-1}$ *for* $k \geq 2$, *let* $T_k$ *denote the first time that* $\mathsf{Loss}_{5,s}^{L_k,2}$ *decreases below* $e^{(-\frac{1}{2}+3.01c_1)\cdot 2B}$. *For all iterations* $t \leq T_k$, *we have the following holds*

1. $\left[\mathbf{Q}_{4,3}^{(t)}\right]_{s,s} + \left[\mathbf{Q}_{4,4}^{(t)}\right]_{s,s}$ *monotonically increases;*

2. *for any sample* $\mathbf{Z}^{2,1}$, *we have* $\mathbf{Attn}_{\mathsf{ans},1\to\mathsf{pred},2}^{(t)} - \mathbf{Attn}_{\mathsf{ans},1\to\mathsf{ans},1}^{(t)} \leq c_1$ *for some sufficiently small constant* $c_1 = \frac{1.005\log d}{2B} > 0$;

3. *for any* $\mathbf{Z}^{2,1}$, *we have* $\mathbf{Attn}_{\mathsf{ans},1\to\mathsf{ans},1}^{(t)} - \mathbf{Attn}_{\mathsf{ans},1\to\mathsf{pred},2}^{(t)} \leq \min\left\{\frac{L_k-1}{L_k}\epsilon_{\mathsf{attn}}^{L_k,2} - c_2, 0\right\}$, *where* $c_2 = \frac{\log d}{4B} > 0$ *is some sufficiently small constant.*

4. *for* $p \in \{3,4\}$, *for* $s' \in \tau(\mathcal{X}) \neq s$, $\left|\left[\mathbf{Q}_{4,p}^{(t)}\right]_{s,s'}\right| \leq O\left(\frac{[\mathbf{Q}_{4,p}^{(t)}]_{s,s}}{d}\right)$ ; *otherwise* $\left[\mathbf{Q}_{4,p}^{(t)}\right]_{s,s'} = 0$.

### I.2.1 Attention and Lambda Preliminaries

**Lemma I.9.** *If Induction I.1 holds for all iterations* $[T_{k-1}, t)$,*then for any sample* $\mathbf{Z}^{L_k,1}$, *we have*

1. $\epsilon_{\mathsf{attn}}^{L_k} \in \left[\frac{4}{3}c_1, 2\epsilon_{\mathsf{attn}}^{L_{k-1}}\right]$;

2. $\mathbf{Attn}_{\mathsf{ans},1\to\mathbf{k}}^{(t)} - \mathbf{Attn}_{\mathsf{ans},1\to\mathbf{k}'}^{(t)} \geq \Omega(1)$ *for any* $\mathbf{k},\mathbf{k}' \in \mathcal{I}^{L_k,1} \setminus \{(\mathsf{ans},1),(\mathsf{pred},2)\}$;

3. $\left|\mathbf{Attn}_{\mathsf{ans},1\to\mathbf{k}}^{(t)} - \mathbf{Attn}_{\mathsf{ans},1\to\mathbf{k}'}^{(t)}\right| \leq \widetilde{O}\left(\frac{1}{d}\right)$ *for any* $\mathbf{k},\mathbf{k}' \in \mathcal{I}^{L_k,1} \setminus \{(\mathsf{ans},1)\}$.

*Proof.* The highest loss occurs when the sample $\mathbf{Z}^{L_k}$ makes the wrong answer highly confused with the correct answer. Thus, we consider the case that $g_\ell \cdot y_0 = g_2 \cdot y_0$ for all $\ell \in [L_k] \setminus \{2\}$ and $y_0 \neq y_1$.

$$\log p_F(\mathbf{Z}_{\mathsf{ans},2,5}|\mathbf{Z}^{(L_k,1)})$$

$$\leq \Theta(1) \cdot \max\left\{ e^{\left(\mathbf{Attn}_{\mathsf{ans},1\to\mathsf{pred},2}^{(t)} + \frac{L_k-1}{L_k}\epsilon_{\mathsf{attn}}^{L_k,2} - \mathbf{Attn}_{\mathsf{ans},1\to\mathsf{ans},1}^{(t)}\right)2B - \left(\mathbf{Attn}_{\mathsf{ans},1\to\mathsf{pred},2}^{(t)} - \frac{1}{L_k}\epsilon_{\mathsf{attn}}^{L_k,2}\right)2B},\right.$$

$$\left. e^{\log d - \left(\mathbf{Attn}_{\mathsf{ans},1\to\mathsf{pred},2}^{(t)} - \frac{1}{L_k}\epsilon_{\mathsf{attn}}^{L_k,2}\right)2B}\right\}$$

$$\leq \Theta\left(e^{\left(\frac{L_k-1}{L_k}\epsilon_{\mathsf{attn}}^{L_k,2} + c_1\right)2B - \left(\mathbf{Attn}_{\mathsf{ans},1\to\mathsf{pred},2}^{(t)} - \frac{1}{L_k}\epsilon_{\mathsf{attn}}^{L_k,2}\right)2B}\right)$$

$$\leq \Theta\left(e^{-\left(\frac{1}{2} - \frac{3}{2}\epsilon_{\mathsf{attn}}^{L_k,2} - c_1\right)2B}\right).$$

Thus, if $\epsilon_{\mathsf{attn}}^{L_k} \leq \frac{4}{3}c_1$, we must have

$$\mathsf{Loss}_{5,s}^{2,2} \leq \Theta\left(e^{\left(-\frac{1}{2}+3\cdot\frac{2c_1}{3}+c_1\right)2B}\right) = \Theta\left(e^{-\left(\frac{1}{2}+3c_1\right)2B}\right),$$

which contradicts the definition of $T_{1,2,2,s}$. Therefore, it must hold that $\epsilon_{\mathsf{attn}}^{L_k} \geq \frac{4}{3}c_1$ for all $t \leq T_k$. $\square$

**Lemma I.10.** *If Induction I.1 holds for all iterations* $[T_{k-1}, t)$, *then given* $\mathbf{Z}^{L_k,1} \notin \mathcal{E}_{L_k,2}$, *we have*

1. *for the prediction* $j_2$, *we have*

$$\Lambda_{5,j_2,r_{g_2\cdot y_1}}^{(t)} = \left(\sum_{\ell\in[L_k]} \mathbb{1}_{g_\ell(y_1)=g_2(y_1)}\mathbf{Attn}_{\mathsf{ans},1\to\mathsf{pred},\ell}^{(t)} - \mathbf{Attn}_{\mathsf{ans},1\to\mathsf{ans},0}^{(t)}\right) \cdot 2B + O\left(\frac{B}{n_y}\right) + O(\delta).$$

2. *for the prediction* $j_2' = \tau\big(g_2(y_0)\big)$, *we have*

$$\Lambda_{5,j_2',r_{g_2\cdot y_0}}^{(t)} = \left(\sum_{\ell\in[L_k]} \mathbb{1}_{g_\ell(y_0)=g_2(y_0)}\mathbf{Attn}_{\mathsf{ans},1\to\mathsf{pred},\ell}^{(t)} - \mathbf{Attn}_{\mathsf{ans},1\to\mathsf{ans},1}^{(t)}\right) \cdot 2B + O\left(\frac{B}{n_y}\right) + O(\delta).$$

3. *for other* $j = g_2(y) \in \tau(\mathcal{Y})$, *noticing that for this case* $y \neq y_0, y_1$, *then we have*

$$\Lambda_{5,j,r_{g_2\cdot y}}^{(t)} = \left(\sum_{\ell\in[L_k]} \mathbb{1}_{g_\ell(y)=g_2(y)}\mathbf{Attn}_{\mathsf{ans},1\to\mathsf{pred},\ell}^{(t)} - \mathbf{Attn}_{\mathsf{ans},1\to\mathsf{ans},1}^{(t)} - \mathbf{Attn}_{\mathsf{ans},1\to\mathsf{ans},0}^{(t)}\right) \cdot 2B$$

$$+ O\Big(\frac{B}{n_y}\Big) + O(\delta).$$

**Lemma I.11.** *If Induction I.1 holds for all iterations $[T_{k-1}, t)$, then given $\mathbf{Z}^{L_k,1} \in \mathcal{E}_{L_k,2}$, we have*

    *1. for the prediction $j_2$, we have*

$$\Lambda^{(t)}_{5,j_2,r_{g_2 \cdot y_1}} = \sum_{\ell \in [L_k]} \mathbb{1}_{g_\ell(y_1) = g_2(y_1)} \mathbf{Attn}^{(t)}_{\mathrm{ans},1 \to \mathrm{pred},\ell} \cdot 2B + O\Big(\frac{B}{n_y}\Big) + O(\delta).$$

    *2. for other $j = \tau(g_2(y)) \in \tau(\mathcal{Y})$ with $y \neq y_1$, then we have*

$$\Lambda^{(t)}_{5,j,r_{g_2 \cdot y}} = \Big( \sum_{\ell \in [L_k]} \mathbb{1}_{g_\ell(y) = g_2(y)} \mathbf{Attn}^{(t)}_{\mathrm{ans},1 \to \mathrm{pred},\ell} - \mathbf{Attn}^{(t)}_{\mathrm{ans},1 \to \mathrm{ans},1} - \mathbf{Attn}^{(t)}_{\mathrm{ans},1 \to \mathrm{ans},0} \Big) \cdot 2B$$

$$+ O\Big(\frac{B}{n_y}\Big) + O(\delta).$$

### I.2.2 Gradient Lemma

**Lemma I.12.** *If Induction I.1 holds for all iterations $t \in [T_{k-1}, T_k)$, given $s \in \tau(\mathcal{X})$, we have*

$$\Big[ -\nabla_{\mathbf{Q}^{(t)}_{4,3}} \mathsf{Loss}^{2,2}_5 \Big]_{s,s} + \Big[ -\nabla_{\mathbf{Q}^{(t)}_{4,4}} \mathsf{Loss}^{2,2}_5 \Big]_{s,s}$$

$$\geq \Omega(B/d) \mathbb{E}\Big[ \epsilon^{L_k}_{\mathrm{attn}} \cdot (1 - \mathbf{logit}^{(t)}_{5,j_2}) \big| \tau(x_1) = s, \mathcal{E}_{L_k,1} \cap \mathcal{E}^c_{L_k,2} \Big].$$

*Proof.* By Induction I.1, Lemma I.10, and Lemma I.11, we have

$$\mathbb{E}\Big[ \mathcal{N}^{(t)}_{s,L_k,i} \Big]$$

$$= \mathbb{E}\Big[ (1 - \mathbf{logit}^{(t)}_{5,j_2}) \cdot \Big( -\sum_{\ell \neq 2} \mathbf{Attn}^{(t)}_{\mathrm{ans},1 \to \mathrm{pred},\ell} \cdot V_{j_2,r_{g_2 \cdot y_1}}(g_\ell) - \mathbf{Attn}^{(t)}_{\mathrm{ans},1 \to \mathrm{ans},0} \cdot V_{j_2,r_{g_2 \cdot y_1}}(y_0)$$

$$+ \big( 1 - \mathbf{Attn}^{(t)}_{\mathrm{ans},1 \to \mathrm{ans},1} - \mathbf{Attn}^{(t)}_{\mathrm{ans},1 \to \mathrm{pred},2} \big) \Lambda^{(t)}_{5,j_2,r_{g_2 \cdot y_1}} \pm \widetilde{O}(\sigma_0) \Big) \pm \widetilde{O}(\delta^q) \Big] \qquad (126)$$

$$\mathbb{1}_{\tau(x_1)=s} \mathbb{1}_{\Lambda^{(t)}_{5,j_2,r_{g_2 \cdot y_1}} \leq B}.$$

Similarly,

$$\mathbb{E}\Big[ \mathcal{N}^{(t)}_{s,L_k,ii} \Big]$$

$$= \mathbb{E}\Big[ \sum_{y \neq y_1} \mathbf{logit}^{(t)}_{5,\tau(g_2(y))} \cdot \mathbf{sReLU}'(\Lambda^{(t)}_{5,\tau(g_2(y)),r_{g_2 \cdot y}})$$

$$\Big( \sum_{\ell \neq 2} \mathbf{Attn}^{(t)}_{\mathrm{ans},1 \to \mathrm{pred},\ell} \cdot V_{\tau(g_2(y)),r_{g_2 \cdot y}}(g_\ell) + \mathbf{Attn}^{(t)}_{\mathrm{ans},1 \to \mathrm{ans},0} \cdot V_{\tau(g_2(y)),r_{g_2 \cdot y}}(y_0)$$

$$- \big( 1 - \mathbf{Attn}^{(t)}_{\mathrm{ans},1 \to \mathrm{ans},1} - \mathbf{Attn}^{(t)}_{\mathrm{ans},1 \to \mathrm{pred},2} \big) \Lambda^{(t)}_{5,\tau(g_2(y)),r_{g_2 \cdot y}} \pm \widetilde{O}(\sigma_0) \Big) \pm \widetilde{O}(\delta^q) \Big] \mathbb{1}_{\tau(x_1)=s}.$$

Consider the event $\mathcal{E}_{L_k,1}$. For $\mathbf{Z}^{L_k,1} \in \mathcal{E}_{L_k,1} \cap \mathcal{E}^c_{L_k,2}$, we have $\Lambda^{(t)}_{5,j_2,r_{g_2 \cdot y_1}} \leq B$, since

$$\Lambda^{(t)}_{5,j_2,r_{g_2 \cdot y_1}} = \sum_{\ell \neq 2} \mathbf{Attn}^{(t)}_{\mathrm{ans},1 \to \mathrm{pred},\ell} \cdot V_{j_2,r_{g_2 \cdot y_1}}(g_\ell) \mathbb{1}_{g_\ell(y_1) = j_2} - \mathbf{Attn}^{(t)}_{\mathrm{ans},1 \to \mathrm{ans},0} \cdot V_{j_2,r_{g_2 \cdot y_1}}(y_0)$$

$$\leq 2 \Big( \mathbf{Attn}^{(t)}_{\mathrm{ans},1 \to \mathrm{pred},2} + (U_k - 1/L_k) \cdot \epsilon^{L_k}_{\mathrm{attn}} \Big) B$$

$$\leq 2 \Big( \frac{1 - \epsilon^{L_k}_{\mathrm{attn}} + c_1}{2} + (U_k - 1/L_k) \cdot \epsilon^{L_k}_{\mathrm{attn}} \Big) B$$

$$= 2 \Big( \frac{1 + c_1}{2} - \big( \frac{1}{2} + 1/L_k - U_k \big) \cdot \epsilon^{L_k}_{\mathrm{attn}} \Big) B.$$

Thus, provided $\epsilon^{L_k}_{\mathrm{attn}} \geq 4c_1/3$ and choosing $U_k = \lfloor 1 + \frac{L_k}{8} \rfloor / L_k$, we indeed have $\Lambda^{(t)}_{5,j_2,r_{g_2 \cdot y_1}} \leq B$.

- For $j = \tau(g_2(y))$ in $\mathcal{N}^{(t)}_{s,L_k,ii}$:
    - $y = y_0 \neq y_1$,
    $$\sum_{\ell \neq 2} \mathbf{Attn}^{(t)}_{\mathsf{ans},1\to\mathsf{pred},\ell} \cdot V_{\tau(g_2(y)),r_{g_2 \cdot y}}(g_\ell) + \mathbf{Attn}^{(t)}_{\mathsf{ans},1\to\mathsf{ans},0} \cdot V_{\tau(g_2(y)),r_{g_2 \cdot y}}(y_0)$$
    $$\geq \sum_{\ell \neq 2} \mathbf{Attn}^{(t)}_{\mathsf{ans},1\to\mathsf{pred},\ell} \cdot V_{\tau(g_2(y)),r_{g_2 \cdot y}}(g_\ell) \mathbb{1}_{g_\ell(y) \neq g_2(y)}$$
    $$\geq -O\left(\frac{B}{n_y}\right) \cdot \left(1 - \mathbf{Attn}^{(t)}_{\mathsf{ans},1\to\mathsf{ans},1} - \mathbf{Attn}^{(t)}_{\mathsf{ans},1\to\mathsf{pred},2}\right). \tag{127}$$

    - $y \neq y_0, y_1$. If there exists at least one $\ell \neq 2$ such that $g_\ell(y) = g_2(y)$, then by cancellation,
    $$\sum_{\ell \neq 2} \mathbf{Attn}^{(t)}_{\mathsf{ans},1\to\mathsf{pred},\ell} \cdot V_{\tau(g_2(y)),r_{g_2 \cdot y}}(g_\ell) + \mathbf{Attn}^{(t)}_{\mathsf{ans},1\to\mathsf{ans},0} \cdot V_{\tau(g_2(y)),r_{g_2 \cdot y}}(y_0)$$
    $$\geq -O\left(\frac{B}{n_y}\right) \cdot \left(1 - \mathbf{Attn}^{(t)}_{\mathsf{ans},1\to\mathsf{ans},1} - \mathbf{Attn}^{(t)}_{\mathsf{ans},1\to\mathsf{pred},2}\right). \tag{128}$$

    Else, pick $y' \neq y, y_1$ so that $\Lambda^{(t)}_{5,\tau(g_2(y')),r_{g_2 \cdot y'}} \geq \Lambda^{(t)}_{5,\tau(g_2(y)),r_{g_2 \cdot y}}$. Then
    $$\mathbf{logit}_{5,\tau(g_2(y))} \leq O\left(\frac{1}{n_y}\right)\left(1 - \mathbf{logit}_{5,j_2}\right),$$
    which implies
    $$\mathbf{logit}^{(t)}_{5,j}\left(\sum_{\ell \neq 2} \mathbf{Attn}^{(t)}_{\mathsf{ans},1\to\mathsf{pred},\ell} \cdot V_{\tau(g_2(y)),r_{g_2 \cdot y}}(g_\ell) + \mathbf{Attn}^{(t)}_{\mathsf{ans},1\to\mathsf{ans},0} \cdot V_{\tau(g_2(y)),r_{g_2 \cdot y}}(y_0)\right)$$
    $$\geq -O\left(\frac{B}{L_k n_y}\right) \cdot \left(1 - \mathbf{logit}_{5,j_2}\right) \cdot \left(1 - \mathbf{Attn}^{(t)}_{\mathsf{ans},1\to\mathsf{ans},1} - \mathbf{Attn}^{(t)}_{\mathsf{ans},1\to\mathsf{pred},2}\right). \tag{129}$$

- For $\mathcal{N}^{(t)}_{s,L_k,i}$:
    - If $\mathbf{Z}^{L_k,1} \in \mathcal{E}_{L_k,1} \cap \mathcal{E}^c_{L_k,2}$, then
    $$-\sum_{\ell \neq 2} \mathbf{Attn}^{(t)}_{\mathsf{ans},1\to\mathsf{pred},\ell} \cdot V_{j_2,r_{g_2 \cdot y_1}}(g_\ell) - \mathbf{Attn}^{(t)}_{\mathsf{ans},1\to\mathsf{ans},0} \cdot V_{j_2,r_{g_2 \cdot y_1}}(y_0)$$
    $$+ \left(1 - \mathbf{Attn}^{(t)}_{\mathsf{ans},1\to\mathsf{ans},1} - \mathbf{Attn}^{(t)}_{\mathsf{ans},1\to\mathsf{pred},2}\right)\Lambda^{(t)}_{5,j_2,r_{g_2 y_1}}$$
    $$\geq -\sum_{\ell \neq 2} \mathbf{Attn}^{(t)}_{\mathsf{ans},1\to\mathsf{pred},\ell} \mathbb{1}_{g_\ell = j_2} \cdot V_{j_2,r_{g_2 \cdot y_1}}(g_\ell) - \mathbf{Attn}^{(t)}_{\mathsf{ans},1\to\mathsf{ans},0} \cdot V_{j_2,r_{g_2 \cdot y_1}}(y_0)$$
    $$+ \left(1 - \mathbf{Attn}^{(t)}_{\mathsf{ans},1\to\mathsf{ans},1} - \mathbf{Attn}^{(t)}_{\mathsf{ans},1\to\mathsf{pred},2}\right)\Lambda^{(t)}_{5,j_2,r_{g_2 y_1}}$$
    $$\geq \left(-\left(U_k - \frac{1}{L_k}\right) \cdot 2B + \Lambda^{(t)}_{5,j_2,r_{g_2 \cdot y_1}}\right) \cdot \epsilon^{L_k}_{\mathsf{attn}} \geq \left(\Lambda^{(t)}_{5,j_2,r_{g_2 \cdot y_1}} - \frac{B}{4}\right) \cdot \epsilon^{L_k}_{\mathsf{attn}}. \tag{130}$$

    - Else, for $\mathbf{Z}^{L_k,1} \in \mathcal{E}^c_{L_k,1} \cup \mathcal{E}_{L_k,2}$, note that having more identical predictions $g_\ell(y_1) = j_2$ or less distraction (i.e., $y_0 = y_1$) increases $\Lambda^{(t)}_{5,j_2,r_{g_2 y_1}}$. Hence
    $$\mathbb{E}\left[(1 - \mathbf{logit}_{5,j_2}) \mid \mathcal{E}^c_{L_k,1} \cup \mathcal{E}_{L_k,2}\right] \leq O(1) \cdot \mathbb{E}\left[(1 - \mathbf{logit}_{5,j_2}) \mid \mathcal{E}_{L_k,1} \cap \mathcal{E}^c_{L_k,2}\right],$$
    and
    $$\left| -\sum_{\ell \neq 2} \mathbf{Attn}^{(t)}_{\mathsf{ans},1\to\mathsf{pred},\ell} \cdot V_{j_2,r_{g_2 \cdot y_1}}(g_\ell) - \mathbf{Attn}^{(t)}_{\mathsf{ans},1\to\mathsf{ans},0} \cdot V_{j_2,r_{g_2 \cdot y_1}}(y_0) \right.$$
    $$\left. + \left(1 - \mathbf{Attn}^{(t)}_{\mathsf{ans},1\to\mathsf{ans},1} - \mathbf{Attn}^{(t)}_{\mathsf{ans},1\to\mathsf{pred},2}\right)\Lambda^{(t)}_{5,j_2,r_{g_2 \cdot y_1}} \right| \leq O(1) \cdot \epsilon^{L_k}_{\mathsf{attn}} \cdot B. \tag{131}$$

Putting the above together, we have

$$\left[-\nabla_{\mathbf{Q}_{4,3}^{(t)}}\mathsf{Loss}_5^{2,2}\right]_{s,s} + \left[-\nabla_{\mathbf{Q}_{4,4}^{(t)}}\mathsf{Loss}_5^{2,2}\right]_{s,s}$$

$$= \mathbb{E}\left[\mathcal{N}_{s,L_k,1}^{(t)}\mathbb{1}_{\mathcal{E}_{L_k,1}\cap\mathcal{E}_{L_k,2}^c}\right] + \mathbb{E}\left[\mathcal{N}_{s,L_k,1}^{(t)}\mathbb{1}_{\Lambda_{5,j_2,r_{g_2\cdot y_1}}^{(t)}\leq B}\mathbb{1}_{\mathcal{E}_{L_k,1}^c\cup\mathcal{E}_{L_k,2}}\right]$$

$$\qquad + \mathbb{E}\left[\mathcal{N}_{s,L_k,2}^{(t)}\right] \pm \widetilde{O}(\sigma_0^q)$$

$$= \mathbb{E}\left[(\mathcal{N}_{s,L_k,1}^{(t)} + \mathcal{N}_{s,L_k,2}^{(t)})\mathbb{1}_{\mathcal{E}_{L_k,1}\cap\mathcal{E}_{L_k,2}^c}\right]$$

$$\qquad + \mathbb{E}\left[\mathcal{N}_{s,L_k,1}^{(t)}\mathbb{1}_{\Lambda_{5,j_2,r_{g_2\cdot y_1}}^{(t)}\leq B}\mathbb{1}_{\mathcal{E}_{L_k,1}^c\cup\mathcal{E}_{L_k,2}}\right] + \mathbb{E}\left[\mathcal{N}_{s,L_k,2}^{(t)}\mathbb{1}_{\mathcal{E}_{L_k,1}^c\cup\mathcal{E}_{L_k,2}}\right] \pm \widetilde{O}(\sigma_0^q).$$

First, by (127), (128), (130),

$$\mathbb{E}\left[(\mathcal{N}_{s,L_k,1}^{(t)} + \mathcal{N}_{s,L_k,2}^{(t)})\mathbb{1}_{\mathcal{E}_{L_k,1}\cap\mathcal{E}_{L_k,2}^c}\right]$$

$$\geq \mathbb{E}\left[(1-\mathbf{logit}_{5,j_2}^{(t)})\left((\Lambda_{5,j_2,r_{g_2\cdot y_1}}^{(t)} - \tfrac{B}{4})\epsilon_{\mathsf{attn}}^{L_k}\right) - (1-\mathbf{logit}_{5,j_2}^{(t)})\left(O(\tfrac{B}{n_y})\epsilon_{\mathsf{attn}}^{L_k} + \max_{y\neq y_1}\Lambda_{5,\tau(g_2(y)),r_{g_2\cdot y}}^{(t)}\right)\right.$$

$$\left. \cdot \mathbb{1}_{\tau(x_1)=s}\mathbb{1}_{\mathcal{E}_{L_k,1}\cap\mathcal{E}_{L_k,2}^c}\right]$$

$$\geq \Omega(B/d)\cdot\mathbb{E}\left[\epsilon_{\mathsf{attn}}^{L_k}\cdot(1-\mathbf{logit}_{5,j_2}^{(t)}) \mid \tau(x_1)=s, \mathcal{E}_{L_k,1}\cap\mathcal{E}_{L_k,2}^c\right].$$

Moreover, by (131),

$$\mathbb{E}\left[\mathcal{N}_{s,L_k,1}^{(t)}\mathbb{1}_{\Lambda_{5,j_2,r_{g_2\cdot y_1}}^{(t)}\leq B/2}\mathbb{1}_{\mathcal{E}_{L_k,1}^c\cup\mathcal{E}_{L_k,2}}\right]$$

$$\geq -\mathbf{Pr}(\mathbf{Z}^{L_k,1}\notin\mathcal{E}_{L_k,1}^c)\cdot O(B/d)\cdot\mathbb{E}\left[\epsilon_{\mathsf{attn}}^{L_k}\cdot(1-\mathbf{logit}_{5,j_2}^{(t)}) \mid \tau(x_1)=s, \mathcal{E}_{L_k,1}^c\cup\mathcal{E}_{L_k,2}\right]$$

$$\geq -\left(O(1/n^{\Omega(1)}) + 1/n_y\right)\cdot O(B/d)\cdot\mathbb{E}\left[\epsilon_{\mathsf{attn}}^{L_k}\cdot(1-\mathbf{logit}_{5,j_2}^{(t)}) \mid \tau(x_1)=s, \mathcal{E}_{L_k,1}\cap\mathcal{E}_{L_k,2}^c\right],$$

and, by (127), (128), (129),

$$\mathbb{E}\left[\mathcal{N}_{s,L_k,2}^{(t)}\mathbb{1}_{\mathcal{E}_{L_k,1}^c\cup\mathcal{E}_{L_k,2}}\right] \geq \tfrac{1}{n_y}\cdot O(B/d)\cdot\mathbb{E}\left[\epsilon_{\mathsf{attn}}^{L_k}\cdot(1-\mathbf{logit}_{5,j_2}^{(t)}) \mid \tau(x_1)=s, \mathcal{E}_{L_k,1}^c\cup\mathcal{E}_{L_k,2}\right].$$

Therefore,

$$\left[-\nabla_{\mathbf{Q}_{4,3}^{(t)}}\mathsf{Loss}_5^{2,2}\right]_{s,s} + \left[-\nabla_{\mathbf{Q}_{4,4}^{(t)}}\mathsf{Loss}_5^{2,2}\right]_{s,s}$$

$$\geq \Omega(B/d)\cdot\mathbb{E}\left[\epsilon_{\mathsf{attn}}^{L_k}\cdot(1-\mathbf{logit}_{5,j_2}^{(t)}) \mid \tau(x_1)=s, \mathcal{E}_{L_k,1}\cap\mathcal{E}_{L_k,2}^c\right].$$

$\square$

**Lemma I.13.** *If Induction I.1 holds for all iterations $t \in [T_{k-1}, T_k)$, given $s \in \tau(\mathcal{X})$, for $[\mathbf{Q}_{4,p}]_{s,s'}$, $p \in \{3,4\}$, $s' \neq s \in \tau(\mathcal{X})$, we have*

$$\left|\left[-\nabla_{\mathbf{Q}_{4,p}^{(t)}}\mathsf{Loss}_5^{2,2}\right]_{s,s'}\right| \leq O(\tfrac{1}{d})\left|\left[-\nabla_{\mathbf{Q}_{4,p}^{(t)}}\mathsf{Loss}_5^{2,2}\right]_{s,s}\right|.$$

### I.2.3 Attention gap is small

**Lemma I.14.** *If Induction I.1 holds for all iterations $t \in [T_{k-1}, T_k)$, then for any sample $\mathbf{Z}^{L_k,1}$, we have*

$$\mathbf{Attn}_{\mathsf{ans},1\to\mathsf{pred},2}^{(t)} - \mathbf{Attn}_{\mathsf{ans},1\to\mathsf{ans},1}^{(t)} \leq c_1,$$

*where $c_1 > 0$ is a small constant.*

*Proof.* The proof follows the same high-level idea as Lemma H.22. The key observation is that for each $\mathbf{Z}^{L_k,1}$, there always exists at least one $y \neq y_1$ such that, for some index $j$ appearing in the

gradient term $\mathcal{N}^{(t)}_{s,L_k,2}$, we have

$$\Lambda^{(t)}_{5,j,r_{g_2 \cdot y}} \geq 2\big(\mathbf{Attn}^{(t)}_{\text{ans},1\to\text{pred},2} - \mathbf{Attn}^{(t)}_{\text{ans},1\to\text{ans},1}\big)B.$$

Let $\widehat{\mathcal{J}}$ denote the set of indices $j$ achieving $\arg\max_{y \notin \{y_1\}} \Lambda^{(t)}_{5,j,r_{g_2 \cdot y}}$, and pick one such index $j' = g_2(y')$. Define $\widetilde{T}$ to be the first time when $\mathbf{Attn}^{(t)}_{\text{ans},1\to\text{pred},2} - \mathbf{Attn}^{(t)}_{\text{ans},1\to\text{ans},1} \geq c_1$. At this time, we have

$$\sum_{j \in \widehat{\mathcal{J}}} \mathbf{logit}^{(\widetilde{T})}_{5,j} = \Big(1 - O(d^{-\Omega(1)})\Big)\Big(1 - \mathbf{logit}^{(\widetilde{T})}_{5,j_2}\Big).$$

Plugging this into the gradient expressions yields

$$\mathcal{N}^{(\widetilde{T})}_{s,3,L_k,i} + \mathcal{N}^{(\widetilde{T})}_{s,3,L_k,ii}$$
$$= \mathbf{Attn}^{(\widetilde{T})}_{\text{ans},1\to\text{pred},2} \cdot \Bigg( \Big(1 - O(d^{-\Omega(1)})\Big)\big(1 - \mathbf{logit}^{(\widetilde{T})}_{5,j_2}\big) \cdot \Big(\Lambda^{(\widetilde{T})}_{5,j',r_{g_2 \cdot y'}} - \Lambda^{(\widetilde{T})}_{5,j_2,r_{g_2 \cdot y_1}}\Big)$$
$$+ O(d^{-\Omega(1)})\big(1 - \mathbf{logit}^{(\widetilde{T})}_{5,j_2}\big) \cdot \Big(V_{5,j_2,r_{g_2 \cdot y_1},2}(g_2) - \Lambda^{(\widetilde{T})}_{5,j_2,r_{g_2 \cdot y_1}}\Big)\Bigg)\mathbb{1}_{\tau(x_1)=s},$$

and similarly

$$\mathcal{N}^{(\widetilde{T})}_{s,4,L_k,i} + \mathcal{N}^{(\widetilde{T})}_{s,4,L_k,ii}$$
$$= \mathbf{Attn}^{(\widetilde{T})}_{\text{ans},1\to\text{pred},2} \cdot \Bigg( \Big(1 - O(d^{-\Omega(1)})\Big)\big(1 - \mathbf{logit}^{(\widetilde{T})}_{5,j_2}\big) \cdot \Big(- V_{j_2',r_{g_2 \cdot y_0}}(y_1) + \Lambda^{(\widetilde{T})}_{5,j',r_{g_2 \cdot y'}} - \Lambda^{(\widetilde{T})}_{5,j_2,r_{g_2 \cdot y_1}}\Big)$$
$$+ O(d^{-\Omega(1)})\big(1 - \mathbf{logit}^{(\widetilde{T})}_{5,j_2}\big) \cdot \Big(V_{5,j_2,r_{g_2 \cdot y_1},2}(y_1) - \Lambda^{(\widetilde{T})}_{5,j_2,r_{g_2 \cdot y_1}}\Big)\Bigg)\mathbb{1}_{\tau(x_1)=s}.$$

Since $\Lambda^{(\widetilde{T})}_{5,j',r_{g_2 \cdot y'}} - \Lambda^{(\widetilde{T})}_{5,j_2,r_{g_2 \cdot y_1}} \leq 0$ and

$$-V_{j_2,r_{g_2 \cdot y_1}}(y_1) - \Lambda^{(\widetilde{T})}_{5,j_2,r_{g_2 \cdot y_1}} + \Lambda^{(\widetilde{T})}_{5,j_2,r_{g_2 \cdot y'}} \geq \Omega(B),$$

we obtain

$$\Big[-\nabla_{\mathbf{Q}^{(\widetilde{T})}_{4,4}}\mathsf{Loss}^{2,2}_5\Big]_{s,s} + \Big[\nabla_{\mathbf{Q}^{(\widetilde{T})}_{4,3}}\mathsf{Loss}^{2,2}_5\Big]_{s,s} \geq \Omega\big(\tfrac{B}{d}\big) \cdot \mathbb{E}\big[1 - \mathbf{logit}^{(\widetilde{T})}_{5,j_2} \mid \tau(x_1) = s\big].$$

Therefore, the quantity $\mathbf{Attn}^{(t)}_{\text{ans},1\to\text{pred},2} - \mathbf{Attn}^{(t)}_{\text{ans},1\to\text{ans},1}$ cannot further increase, and we conclude that

$$\mathbf{Attn}^{(t)}_{\text{ans},1\to\text{pred},2} - \mathbf{Attn}^{(t)}_{\text{ans},1\to\text{ans},1} \leq c_1.$$

$\square$

**Lemma I.15.** *If Induction I.1 holds for all iterations $t \in [T_{k-1}, T_k)$, then for any sample $\mathbf{Z}^{L_k,1}$, we have*

$$\mathbf{Attn}^{(t)}_{\text{ans},1\to\text{ans},1} - \mathbf{Attn}^{(t)}_{\text{ans},1\to\text{pred},2} \leq \frac{L_k - 1}{L_k}\epsilon^{L_k}_{\text{attn}} - c_2,$$

*where $c_2 = \frac{\log d}{2B} > 0$ is a small constant.*

*Proof.* The argument parallels Lemma H.34, but in the recursive setting we first note that for every $\mathbf{Z}^{L_k,1} \in \mathcal{E}^c_{L_k,2}$,

$$\Lambda^{(t)}_{5,\tau(g_2(y_0)),r_{g_2 \cdot y_0}} \leq \mathbf{Attn}^{(t)}_{\text{ans},1\to\text{pred},2} - \mathbf{Attn}^{(t)}_{\text{ans},1\to\text{ans},1} + \frac{L_k - 1}{L_k}\epsilon^{L_k}_{\text{attn}}.$$

Let $\widetilde{T}$ be the first time such that

$$\mathbb{E}\Big[\mathbf{Attn}^{(t)}_{\text{ans},1\to\text{pred},2} - \mathbf{Attn}^{(t)}_{\text{ans},1\to\text{ans},1} + \frac{L_k - 1}{L_k}\epsilon^{L_k}_{\text{attn}} \mid \tau(x_1) = s\Big] \leq \frac{\log d}{4.02\,B}.$$

Then, for any $y$,

$$\Lambda^{(\widetilde{T})}_{5,\tau(g_2(y)),\,r_{g_2\cdot y}}$$

$$= 2\big(\mathbf{Attn}^{(\widetilde{T})}_{\mathsf{ans},1\to\mathsf{pred},2} + \sum_{\ell\neq 2}\mathbf{Attn}^{(\widetilde{T})}_{\mathsf{ans},1\to\mathsf{pred},\ell}\,\mathbb{1}_{g_\ell(y)=g_2(y)} - \mathbf{Attn}^{(\widetilde{T})}_{\mathsf{ans},1\to\mathsf{ans},0}\big)\,B \;\leq\; \frac{\log d}{2.01}.$$

Hence the corresponding wrong-class logit mass is small:

$$\mathbf{logit}^{(1)}_{5,j_2} \;\leq\; O\big(d^{-1.01/2.01}\big)\big(1-\mathbf{logit}^{(t)}_{5,j_2}\big).$$

Plugging this into the gradient decomposition, we obtain

$$\mathcal{N}^{(\widetilde{T})}_{s,3,L_k,i} + \mathcal{N}^{(\widetilde{T})}_{s,3,L_k,ii}$$

$$= \mathbf{Attn}^{(\widetilde{T})}_{\mathsf{ans},1\to\mathsf{pred},2}\Big(\big(1-\mathbf{logit}^{(\widetilde{T})}_{5,j_2}\big)\big(V_{5,j_2,r_{g_2\cdot y_1},2}(g_2)-\Lambda^{(\widetilde{T})}_{5,j_2,r_{g_2\cdot y_1}}\big)$$

$$\qquad - O\big(d^{-1.01/2.01}\big)\big(1-\mathbf{logit}^{(\widetilde{T})}_{5,j_2}\big)\sum_{y\neq y_0}\big(V_{5,j_2,r_{g_2\cdot y},2}(g_2)-\Lambda^{(\widetilde{T})}_{5,j_2,r_{g_2\cdot y}}\big)\Big)\,\mathbb{1}_{\tau(x_1)=s}$$

$$\geq \mathbf{Attn}^{(\widetilde{T})}_{\mathsf{ans},1\to\mathsf{pred},2}\cdot\Omega(1)\cdot\big(1-\mathbf{logit}^{(\widetilde{T})}_{5,j_2}\big)\cdot\big(V_{5,j_2,r_{g_2\cdot y_1},2}(g_2)-\Lambda^{(\widetilde{T})}_{5,j_2,r_{g_2\cdot y_1}}\big)\,\mathbb{1}_{\tau(x_1)=s},$$

and

$$\mathcal{N}^{(\widetilde{T})}_{s,4,L_k,i} + \mathcal{N}^{(\widetilde{T})}_{s,4,L_k,ii}$$

$$= \mathbf{Attn}^{(\widetilde{T})}_{\mathsf{ans},1\to\mathsf{pred},2}\Big(\big(1-\mathbf{logit}^{(\widetilde{T})}_{5,j_2}\big)\big(V_{5,j_2,r_{g_2\cdot y_1},2}(y_1)-\Lambda^{(\widetilde{T})}_{5,j_2,r_{g_2\cdot y_1}}\big)$$

$$\qquad - O\big(d^{-1.01/2.01}\big)\big(1-\mathbf{logit}^{(\widetilde{T})}_{5,j_2}\big)\sum_{y\neq y_0}\big(V_{5,j_2,r_{g_2\cdot y},2}(y_1)-\Lambda^{(\widetilde{T})}_{5,j_2,r_{g_2\cdot y}}\big)\Big)\,\mathbb{1}_{\tau(x_1)=s}$$

$$\leq -\mathbf{Attn}^{(\widetilde{T})}_{\mathsf{ans},1\to\mathsf{pred},2}\cdot\Omega(1)\cdot\big(1-\mathbf{logit}^{(\widetilde{T})}_{5,j_2}\big)\cdot\Lambda^{(\widetilde{T})}_{5,j_2,r_{g_2\cdot y_1}}\,\mathbb{1}_{\tau(x_1)=s}.$$

Therefore,

$$\big[-\nabla_{\mathbf{Q}^{(\widetilde{T})}_{4,3}}\mathsf{Loss}^{2,2}_5\big]_{s,s} \;\geq\; \Omega\Big(\frac{B}{d}\Big)\cdot\mathbb{E}\big[1-\mathbf{logit}^{(\widetilde{T})}_{5,j_2}\,\big|\,\tau(x_1)=s,\,\mathcal{E}^c_{L_k,2}\big], \tag{132}$$

$$\big[-\nabla_{\mathbf{Q}^{(\widetilde{T})}_{4,4}}\mathsf{Loss}^{2,2}_5\big]_{s,s} \;\leq\; -\Omega\Big(\frac{B}{d}\Big)\cdot\mathbb{E}\big[1-\mathbf{logit}^{(\widetilde{T})}_{5,j_2}\,\big|\,\tau(x_1)=s,\,\mathcal{E}^c_{L_k,2}\big]. \tag{133}$$

Finally, recall that

$$\mathbf{Attn}^{(t)}_{\mathsf{ans},1\to\mathsf{pred},2} + \mathbf{Attn}^{(t)}_{\mathsf{ans},1\to\mathsf{pred},1} - \mathbf{Attn}^{(t)}_{\mathsf{ans},1\to\mathsf{ans},1} \;=\; \frac{e^{[\mathbf{Q}^{(t)}_{4,3}]_{s,s}} - e^{[\mathbf{Q}^{(t)}_{4,4}]_{s,s}} - e^{\widetilde{O}(1/d)}}{e^{[\mathbf{Q}^{(t)}_{4,3}]_{s,s}} + e^{[\mathbf{Q}^{(t)}_{4,4}]_{s,s}} + e^{\widetilde{O}(1/d)}}.$$

Combining the two gradient inequalities shows that the quantity

$$\mathbf{Attn}^{(t)}_{\mathsf{ans},1\to\mathsf{pred},2} + \frac{L_k-1}{L_k}\,\epsilon^{L_k}_{\mathsf{attn}} - \mathbf{Attn}^{(t)}_{\mathsf{ans},1\to\mathsf{ans},1}$$

must strictly decrease at time $\widetilde{T}+1$. $\qquad\qquad\square$

### I.2.4   At the end of stage

**Lemma I.16** (At the end of stage). *Induction I.1 holds for all iterations $T_{k-1} < t \leq T_k = O(\frac{\mathsf{poly}(d)}{\eta B})$. At the end of stage $k$, we have*

(a) *Attention concentration: $\epsilon^{L_k}_{\mathsf{attn}} \leq 5.04 c_1$ for some small constant $c_1 = \frac{1.005\log d}{B} > 0$;*

(b) *Loss convergence: $\mathsf{Loss}^{L_k,2}_{F^{(T_{k-1})},5} \leq e^{(-\frac{1}{2}+3.01c_1)2B} = \frac{1}{\mathsf{poly}d}$.*

*Proof.* By Lemmas I.9 and I.12, the diagonal entries $[\mathbf{Q}_{4,3}^{(t)}]_{s,s}$ and $[\mathbf{Q}_{4,4}^{(t)}]_{s,s}$ keep increasing until the total wrong-attention mass satisfies $\epsilon_{\text{attn}} \geq \frac{4c_1}{3}$. Combining Lemmas I.12 to I.15, there exists a stopping time

$$T_k = O\left(\frac{\text{poly}(d)}{\eta B}\right),$$

such that Induction I.1 holds for all $t \in [T_1, T_k)$.

To obtain an upper bound on $\epsilon_{\text{attn}}^{L_k}$ at the end of stage $k$, it suffices to work on the main event $\mathcal{E}_{L_k,3}$.

*Case 1: $L_k = O(1)$.* For each $y \in \mathcal{Y}$, at most one index $\ell \neq 2$ can satisfy $g_\ell(y) = g_2(y)$, and thus the number of collisions per $y$ is uniformly bounded by $W_k$. Hence,

$\log p_F(Z_{\text{ans},2,5} \mid Z^{(2,1)})$

$$\geq \Theta(1) \cdot \exp\left(\log d - \left(\mathbf{Attn}_{\text{ans},1\to\text{pred},2}^{(t)} + \sum_{\ell \neq 2} \mathbf{Attn}_{\text{ans},1\to\text{pred},\ell}^{(t)} \mathbb{1}_{g_\ell(y_1)=g_2(y_1)} - \mathbf{Attn}_{\text{ans},1\to\text{ans},0}^{(t)}\right) 2B\right)$$

$$\geq \Theta(1) \cdot \exp\left(\log d - \left(\mathbf{Attn}_{\text{ans},1\to\text{pred},2}^{(t)} + (1/L_k - 1/L_k)\,\epsilon_{\text{attn}}^{L_k}\right) 2B\right)$$

$$\geq \Theta\left(\exp\left(-\left(\tfrac{1}{2} + \tfrac{c_1}{2} - \tfrac{\epsilon_{\text{attn}}^{L_k}}{2} - \tfrac{\log d}{B}\right) 2B\right)\right),$$

where the last line uses $\mathbf{Attn}_{\text{ans},1\to\text{pred},2}^{(t)} \leq \frac{1}{2}(1 - \epsilon_{\text{attn}}^{L_k} + c_1) = \frac{1+c_1}{2} - \frac{\epsilon_{\text{attn}}^{L_k}}{2}$.

*Case 2: $L_k = \omega(1)$.* If $L_k \leq n_y \log n_y$ then $W_k/L_k = O(1/L_k)$; otherwise $W_k/L_k = O(1/n_y)$. In either regime $W_k/L_k = o(1)$, and we similarly obtain

$$\log p_F(Z_{\text{ans},2,5} \mid Z^{(2,1)}) \geq \Theta(1) \cdot \exp\left(\log d - \left(\mathbf{Attn}_{\text{ans},1\to\text{pred},2}^{(t)} + (o(1) - 1/L_k)\,\epsilon_{\text{attn}}^{L_k}\right) 2B\right)$$

$$\geq \Theta\left(\exp\left(-\left(\tfrac{1}{2} + \tfrac{c_1}{2} - \tfrac{\epsilon_{\text{attn}}^{L_k}}{2} - \tfrac{\log d}{B}\right) 2B\right)\right).$$

At convergence we therefore have

$$\tfrac{1}{2} + \tfrac{c_1}{2} - \tfrac{\epsilon_{\text{attn}}^{L_k}}{2} - \tfrac{\log d}{B} \geq \tfrac{1}{2} - 3.01\,c_1,$$

which rearranges to

$$\epsilon_{\text{attn}}^{L_k} \leq 2(3.02c_1 + 0.5c_1 - c_1) = 5.04c_1.$$

In particular, since $B/\log d$ can be taken as a sufficiently large constant, we obtain the clean stage-end bound

$$\epsilon_{\text{attn}}^{L_k} = O(c_1).$$

$\square$

## I.3   Proof of Main Theorem

**Theorem I.1** (Recursive self-training (Restatement)). *Assume the distribution $\mathcal{D}^L$ induced from LEGO$(\mathcal{X}, \mathcal{G}, \mathcal{Y})$ satisfies Assumption 3.1, 3.2 and 5.1, and assume the transformer network satisfies Assumption 3.3, C.1, C.2, and $n_y < m \ll \log^2 d$. Then for any $1 \leq k < \log_2 |\mathcal{X}|$, the transformer $F^{(T_k)}$ trained via Algorithm 2 up to length $L_k = 2^k$ and $T_k = O(\frac{\text{poly}(d)}{\eta})$ satisfies: $F^{(T_k)}$ is able to solve $\mathcal{T}^{L_{k+1}}$ with $L_{k+1} = 2^{k+1}$, i.e.,*

$$\text{Acc}_{L_{k+1}}(F^{(T_k)}) = 1 - O(1/\text{poly}(d)).$$

*Proof.* By Lemma I.16, at the time $T_k$, we have $\epsilon_{\text{attn}}^{L_k} \leq 5.04c_1$, combining with Induction I.1, non-diagonal entry of $\mathbf{Q}_{p,q}$ remains close to 0, thus moving to the next stage, we have $\epsilon_{\text{attn}}^{L_{k+1},\ell} \leq 4\epsilon_{\text{attn}}^{L_k} \leq 20.16c_1$ (notice that $20.16c_1$ could be sufficiently small e.g., 0.01 since we can set a reasonably large B) for all $\ell < L_{k+1}$. Moreover, $\Delta^{L_{k+1},\ell} \leq \Delta^{L_k,1}$. Hence, we have

$$\mathbb{E}_{Z^{L_{k+1}} \sim \mathcal{D}^{L_{k+1}}}\left[\mathbb{E}_{\widehat{Z}_{\text{ans},\ell+1} \sim p_F^{(T_k)}(\cdot \mid Z^{L_{k+1},\ell})}\left[\mathbb{1}_{\{\widehat{Z}_{\text{ans},\ell+1} \neq Z_{\text{ans},\ell+1}\}}\right]\right]$$

$$\leq O(1) \cdot \mathbb{E}_{Z^{L_{k+1}} \sim \mathcal{D}^{L_{k+1}}} \left[ 1 - \mathbf{logit}_{5,\tau(\mathbf{Z}_{\mathsf{ans},\ell+1,5})} \right]$$

$$\leq O(1) \cdot e^{\left( -(\frac{1}{2} - \epsilon_{\mathsf{attn}}^{L_{k+1}} - (\epsilon_{\mathsf{attn}}^{L_{k+1}} - c_2)/2 - \epsilon_{\mathsf{attn}}^{L_{k+1}}) \right) \cdot 2B}$$

$$\leq O(1) \cdot e^{\left( -\frac{1}{2} - \frac{\log d}{4B} + \frac{5}{2} \cdot 0.01 \right) 2B} = O\left( \frac{1}{\mathsf{poly} d} \right).$$

Thus $\mathrm{Acc}_{L_{k+1}}(F^{T_k}) \geq 1 - O\left( \frac{1}{\mathsf{poly} d} \right).$ $\qquad\qquad\qquad\qquad\qquad\square$

