# OpenReview forum: "Transformers Provably Learn Chain-of-Thought Reasoning with Length Generalization"
_NeurIPS.cc/2025/Conference — NeurIPS 2025 poster_

### Official Review · Reviewer_2b2M · 2025-06-19

**Clarity:** 4
**Significance:** 3
**Originality:** 3
**Rating:** 5
**Confidence:** 3

**Summary:**

The paper theoretically studies length generalization in trained transformers for LEGO tasks (i.e., state tracking under $G$-actions). The studied transformer model consists of a constrained softmax attention and smoothed ReLU feedforward layer, without positional encoding. The model is trained on CoT loss via a curriculum of one-step and two-step reasoning. It is shown that for simply transitive actions, the CoT model trained for $poly(d)$ time can generalize up to length $poly(d)$ where $d$ is problem size. Moreover, when $G$ is the symmetric group on the state space, a bootstrap self-training curriculum of progressively doubling length can generalize to longer tasks. This also constitutes a novel optimization guarantee for constant depth transformers beyond the $TC^0$ class.

**Questions:**

See Weaknesses

**Ethical Concerns:**

["NO or VERY MINOR ethics concerns only"]

**Final Justification:**

The paper is a solid contribution to understanding length generalization on the synthetic but illuminating LEGO task. The setup and optimization analysis, while necessarily stylized due to the complexity of the transformer model, also seems to be novel compared to existing works.

**Limitations:**

Yes

**Quality:**

3

**Strengths And Weaknesses:**

**Strengths**

The paper rigorously studies a problem of crucial importance, namely the acquisition mechanism of length generalization in CoT transformers. The considered LEGO task is an empirically well-studied synthetic task, and the one-layer transformer model is also very natural (e.g., incorporates a smooth ReLU feedforward layer, compared to other works which fix or discard the feedforward layer). While there are some structural assumptions and a curriculum learning setup is required to separate the parameters dynamics, I think these are still quite weak for optimization guarantees. The paper is overall easy to read and obtains solid generalization guarantees which displays the potential of length generalization from a theoretical perspective.

**Weaknesses/Questions**

The settings of the two results are restricted to simply transitive actions and action of the full symmetric group for Theorems 4.1 and 4.2, respectively. While the difficulty of analyzing even these settings is evident from the laborious treatment in the appendix, I am curious as to where the technical difficulty arises when attempting to study more general actions, and what we may achieve with future works.

* Why is the assumption of simple transitivity necessary for Theorem 4.1?
* The symmetric group assumption of Theorem 4.2 is rather restrictive in that it necessitates $|\mathcal{Y}| < \log\log d$ so that $|G|$ is logarithmic. With Cayley's theorem in mind, can this be relaxed/generalized to (certain) subgroups of symmetric groups?
* The heading of Section 4.2 mentions non-solvable groups. While it is true that large symmetric groups are non-solvable, unless the theory actually leverages the group-theoretic notion of solvability, it would be better to just say "symmetric group" for clarity.
* Is there a setting in the LEGO setup where length generalization provably fails? (e.g., lower bounds without CoT which have been demonstrated in e.g. the parity papers)

Some typos:
"action group action" in line 242, "atetntion head" in line 301

---

> ### Author Rebuttal · Authors · 2025-07-31
>
> We thank the reviewer for the positive comments and thoughtful questions. Below, we provide point-by-point response to the main comments.
>
> **Q1.** Why is the assumption of simple transitivity?
>
> **A1.** The assumption of simple transitivity can be thought of as a natural generalization of cyclic group actions. Our analysis only requires that for each pair of $y_1, y_2$, there exists a unique group operation $g$ such that $g⋅y_1=y_2$​. In our context, this assumption allows us to study group-based reasoning tasks while keeping the structure simple for analysis.
>
>
> **Q2.** Can the symmetric group assumption in Theorem 4.2 be relaxed/generalized?
>
> **A2**. This is a great question. In fact, our proof can be directly generalized to certain subsets of the symmetric group that satisfy the following condition: for any pair of elements $y, y^{'}$ in the domain, the number of group elements mapping $y$ to $y’$ is a fixed constant. This includes subsets of symmetric groups such as the groups of all k-swaps in $S_n$, for $k \leq n$. These subsets are not necessarily groups (they can be conjugacy classes) but they provide exactly the regularity we need for successful feature learning. If this structure is removed entirely, the distribution of $(g,y)$ pairs becomes imbalanced, which can harm the learning of both single-step reasoning and context-length generalization. While we agree it would be very interesting to extend the analysis to other groups, it is beyond the current scope of our work.
>
> **Q3.** It would be better to just say "symmetric group" for clarity.
>
> **A3**. Thank you for your helpful suggestions! We will revise the wording in the revision.
>
> **Q4.** Is there a setting in the LEGO setup where length generalization provably fails?
>
> **A4.** Our theory links length generalization to the model’s ability to perform in-context retrieval: as context grows, the attention mechanism must still assign enough mass to every relevant clause. Two key factors determine whether generalization can succeed:
> 1. The peak attention attained by the relevant clauses within training context length;
> 2. The rate at which attention is diluted by irrelevant clauses as context length increases.
>
> In the case of a simply transitive group action, the learned features are sufficiently “pure,” enabling sharply focused attention and yielding a polynomial bound on generalization length. In contrast, in the symmetric group action setting, the learned features do not have a similarly strong guarantee, limiting the length generalization to only a constant multiple of the training length.  While our exact theoretical predictions may not transfer directly to all LEGO settings due to differences in task structure and objective, we hope our analysis offers useful insight into when and why length generalization can succeed or fail, especially in realistic transformer-based reasoning scenarios.
>
> ---
> We thank the reviewer again for your comments. We are more than happy to answer your further questions.

---

> > ### Comment · Reviewer_2b2M · 2025-08-03
> >
> > Thank you for the detailed reply. I will maintain my high rating of the contribution.

---

### Official Review · Reviewer_vE9B · 2025-06-29

**Clarity:** 3
**Significance:** 2
**Originality:** 2
**Rating:** 3
**Confidence:** 3

**Summary:**

The paper proves the learning/generalization of CoT for a 1-layer transformer for LEGO, a reasoning task defined by the authors which involves performing a limited set of group operation on state spaces. The paper also proposes a recursive self-training scheme that enables CoT length extrapolation on the LEGO task.

**Questions:**

None

**Ethical Concerns:**

["NO or VERY MINOR ethics concerns only"]

**Final Justification:**

I will maintain my rating as my concerns are not fully addressed (overclaim in title, weaker evaluation section and the issue of no position embeddings).

**Quality:**

2

**Strengths And Weaknesses:**

Strength:
1. The paper is in general written in a clean manner with clear math annotation usage.
2. There is some novelty in proofing CoT for a 1-layer transformer for LEGO, a reasoning task defined by the authors which involves performing a limited set of group operation on state spaces. The recursive self-training also seems an interesting idea that can be applied to some other reasoning problem.

Weakness:
1. The paper's title seems to suggest it proves for all reasoning tasks but in reality the paper only tackles a limited subset of the reasoning tasks which only include a limited set of group transformations on a state space.
2. The evaluation section is weak. There is no detailed description of the datasets (how big is the training set? some example data), the training setup (how many GPUs? trained for how long). There is also little analysis of the results.
3. Sometimes assumptions are stated but without enough explanation. For example, the authors stated that "we use no positional encoding (NoPE) in our attention mechanism" on line 202, but never explained in details why this is the case. Modern LLMs all use positional embeddings which makes the proof not applicable to them.

---

> ### Author Rebuttal · Authors · 2025-07-31
>
> We thank the reviewer for the time and thoughtful comments on our work. Below we address your concerns.
>
> **Q1.**  The paper's title seems to suggest it proves for all reasoning tasks but in reality the paper only tackles a limited subset of the reasoning tasks which only include a limited set of group transformations on a state space.
>
> **A1.**  We would like to respectfully clarify that our title is not an overclaim in the context of existing theoretical work on reasoning with transformers.
> 1. State tracking has been widely adopted as a standard synthetic testbed for evaluating the reasoning abilities of language models. The seminal work [15] established foundational expressivity results for transformers in algorithmic reasoning and provided comprehensive empirical evidence across architectures, from RNNs to transformers, on tasks ranging from Dyck languages to non-solvable groups. It showed that sufficiently deep transformers can solve most group word problems within the training length. Subsequent studies [12–13] further demonstrated the expressivity gap: constant-depth transformers without CoT cannot solve problems beyond TC0, whereas adding a scratchpad (analogous to CoT) enables transformers to solve NC1-complete problems such as S5 [1,2,15,16]. Our work bridges the gap between theory and empirical observations. We go beyond prior results by providing a formal proof that transformers can learn a length-generalizable CoT solution, overcoming the expressivity limitations of the base architecture. This confirms that transformers can learn solutions that are provably more powerful than what the architecture can express without CoT.
> 2. To our knowledge, no prior theoretical paper on CoT has established guarantees for **all** reasoning tasks [1-7]. From the perspective of optimization analysis, existing theoretical studies on CoT [8-10] have exclusively addressed specific, simple tasks within TC0, such as parity or linear regression.  In contrast, our work represents significant advances: 1). We provide the first rigorous analysis of transformer training dynamics for a broader and more challenging class of CoT-style reasoning tasks. Specifically, we study tasks beyond parity within TC0 (simply transitive setting) as well as tasks within NC1 (symmetry group setting); 2). We are the first to establish convergence guarantees that demonstrate not only the transformers' ability to solve these tasks but, crucially, their capacity for **length generalization**.
>
> **Q2.** The evaluation section is weak
>
> **A2**. We would like to emphasize that the empirical part primarily serves as numerical verification of our theoretical results, specifically illustrating length generalization in the cyclic case. Due to space constraints, we omitted detailed dataset descriptions; however, the synthetic data and loss function strictly follow the LEGO structure analyzed theoretically in our paper. Basic setup about network architectures and optimizers is briefly summarized in the paper. Other detailed training setups mentioned by the reviewer, which are less directly related to our theoretical focus, have been intentionally omitted to maintain clarity and conciseness.
>
> **Q3.** Sometimes assumptions are stated, but without enough explanation, For example, the authors stated that "we use no positional encoding (NoPE) in our attention mechanism"
>
> **A3**. We thank the reviewer for raising the concern regarding assumptions stated without sufficient explanation. Specifically,  for the use of "no positional encoding (NoPE)," we note that it has been widely observed empirically that standard positional embeddings inherently fail to achieve length generalization. This limitation arises because traditional positional embeddings implicitly introduce approximate biases that typically favor local positions and thus fundamentally hinder extrapolation beyond the training sequence length. In contrast, our theoretical analysis supports an alternative approach and suggests an alternate transformer design with no positional embedding that provably achieves length generalization, thereby offering insights and guidance for current empirical research. Furthermore, our theory implicitly reveals that the inability of existing networks to generalize to unseen lengths may stem from the presence of positional embeddings, thereby identifying a potential underlying reason why current settings fail in this regard. In the revision, we will carefully include additional discussion and remarks to explicitly clarify these assumptions and the rationale behind our design choices.
>
> ---
>
> We thank the reviewer again for your comments. We hope that our responses resolved your concerns. If so, we wonder if the reviewer could kindly consider to increase your score. Certainly, we are more than happy to answer your further questions.
>
> ---
>
> References:
>
> [1] Feng et al. Towards revealing the mystery behind chain of thought: a theoretical perspective.
>
> [2] Li et al. Chain of thought empowers transformers to solve inherently serial problems.
>
> [3] Merrill et al. The expressive power of transformers with chain of thought.
>
> [4] Abbe., et al. How far can transformers reason? the globality barrier and inductive scratchpad.
>
> [5] Kim et al. Metastable dynamics of chain-of-thought reasoning: Provable benefits of search, rl and distillation.
>
> [6] Huang et al. A formal framework for understanding length generalization in transformers.
>
> [7] Golowich et al. The role of sparsity for length generalization in transformers.
>
> [8] Huang et al. Transformers learn to implement multi-step  gradient descent with chain of thought.
>
> [9] Kim et al. Transformers provably solve parity efficiently with a chain of thought.
>
> [10] Wen et al. From sparse dependence to sparse attention: unveiling how chain-of-thought enhances transformer sample efficiency.
>
> [11] Kim et al. Entity tracking in language models.
>
> [12] Merrill et al. A Little Depth Goes a Long Way: The Expressive Power of Log-Depth Transformers.
>
> [13] Merrill et al. The Illusion of State in State-Space Models.
>
> [14] Zhang et al. Unveiling Transformers with LEGO: a synthetic reasoning task.
>
> [15] Liu et al. Transformers learn shortcuts to automata.
>
> [16] Wen et al. RNNs are not Transformers (Yet): The Key Bottleneck on In-context Retrieval.

---

> > ### Comment · Reviewer_vE9B · 2025-08-07
> >
> > Thanks for the rebuttal. I have no issues with works that only focus on a subset of reasoning tasks, but it's better to reflect that in the title. There are many natural language based reasoning tasks that are fuzzy and not exactly state tracking. While I agree that positional embedding reduces generalization beyond context length, it bring many other benefits that are needed in today's LLM tasks. Thus I think omitting it is not helpful in increasing the impact of this work. I decide to maintain my score.

---

> ### Author Response · Authors · 2025-08-04
>
> Dear Reviewer vE9B,
>
> As the author-reviewer discussion period will end soon, we would like to check whether our responses have properly addressed your concerns? If so, could you please kindly consider increasing your initial score accordingly? Certainly, we are more than happy to answer your further questions.
>
> Thank you for your time and effort in reviewing our work!
>
> Best Regards,
>
> Authors

---

> ### Author Response · Authors · 2025-08-06
> **Your prompt response is highly appreciated**
>
> Dear Reviewer vE9B,
>
>
> As the author-reviewer discussion period ends soon, we would greatly appreciate it if your could check our response promptly. If our response resolves your concerns, we kindly ask you to consider raising the rating of our work. We are also more than happy to answer your further questions. Thank you very much for your time and efforts!
>
> Many thanks,
>
> Authors

---

### Official Review · Reviewer_kcpH · 2025-06-30

**Clarity:** 4
**Significance:** 3
**Originality:** 3
**Rating:** 5
**Confidence:** 3

**Summary:**

The paper provides a theoretical analysis of the ability of transformers to learn state-tracking problems belonging to $\text{NC}^1$ by learning to produce the intermediate reasoning steps, as in the popularly used Chain of Thought setting. This means that sequential prediction methods such as CoT provably extend the capabilities of transformers, unless $\text{NC}^1 = \text{TC}^0$, which is an open problem but is popularly believed to be false. The proofs do make some assumptions on the dimensionality and structure of the transformer, which means they do not immediately apply to the popular transformer architectures as used by practitioners. The scope of the result nonetheless appears to be significant, since it brings seminal work on the expressivity of transformers closer to the CoT paradigm typically of use today.

**Questions:**

Suggestions:
- Please enhance the legibility of Figure 1 substantially. The text is impossible to read when printed out.
- Please add a non-CoT baseline to your experiment to give some credence to the impact on downstream performance implied by your theory.
- Some typos:
  - P.3, Theorem 1.2, second bullet point: "from its own outputs**, .**"
  - P.4, example of LEGO sentence: in answers, $x_L$ should likely be $x_{L'}$
  - P.4, in Eqn (2), $L$ should be $L'$ (per the preceding sentence)
  - P.9, "Note that **in by**"
  - P.9, Experiments, "transformer **with attention with two atetnion heads**"

Questions:
- If I understood your construction on P.5-6 correctly, your transformer in effect uses the same projection matrix $Q$  for the queries and keys, while not using a projection at all for the values. Is that correct? If so, why is this not explicitly listed as an assumption in the paper, since it does not correspond to the more general transformer architecture?
- Assumption 3.4 appears to be very strong. Could you help me build some intuition for why this is necessary? In particular, why $(p, q) \notin \{(4, 3), (4, 4)\}$?
- My understanding is that your clause-wise embedding structure is what allows the model to disentangle the differing meanings of the tokens without positional encodings. Is that correct? If so, how well do you expect that your results would carry over to a RoPE (or other PE) setting in which the clauses are **not** hard-coded to be separated, as is the typical case in language modeling?

**Ethical Concerns:**

["NO or VERY MINOR ethics concerns only"]

**Final Justification:**

My primary concerns revolved around the original paper being a bit unclear about (1) how the assumptions affected the applicability of the theoretical results to real-world models, and (2) why those assumptions were needed in the first place. I think the authors' have substantially improved the paper in both of those respects during the rebuttal and am confident that the final paper will be clearer.

I also had some concerns about the empirical results being presented. These have been extended and improved upon (in scope) during the rebuttal; I still think they are a bit too small and naive to really say much about how the theory here extends to practice, but I do not assign much weight to this as I do not believe it to be a realistic goal that every theoretical contribution to the literature should also have to include experiments with state-of-the-art models (as long as the *theoretical* connection between the two is clear). In particular, I do not share vE9B's apparent sentiment that the experiments are grounds to reject the paper on.

I therefore update my score to 5 - Accept. However, I note to the AC that this branch of theoretical modeling of transformers is not my main research area, and so I have limited my confidence score to 3.

**Limitations:**

Please see questions above.

**Quality:**

4

**Strengths And Weaknesses:**

Strengths:
- The paper is fairly readable. There are a few minor typos, but nothing major.
- As far as I can tell, the main results are quite exciting.
- By basing their study off a task that has also been used to study similar aspects of transformers empirically in the past, the paper does provide some linkage to the broader literature.

Weaknesses:
- While the limitations and assumptions are acknowledged well in the paper, their downstream impact on the generalizability of the results is not clear to me.
- Because of the clause-embedding mechanism, and lack of other positional encoding, it is unclear to me whether these results actually correspond as well to CoT as the authors claim, in which the structure of the sequence (and thus the attention mechanism) is substantially different.
- The experiment could benefit from a non-CoT baseline to display the difference in length generalization empirically. (I realize empirical performance is not the subject being studied here, but it would give some sense of the practical implications of the very theoretical results.)

---

> ### Author Rebuttal · Authors · 2025-07-31
>
> We thank the reviewer for providing the valuable feedback! We will enhance the readability of the figure and fix the typo in the revision. Below we address your concerns.
>
> **Q1.** While the limitations and assumptions are acknowledged, their impact on generalizability is unclear
>
> **A1**. We thank the reviewer for raising this general yet important point. Indeed, as in most theoretical works, certain simplifying assumptions are necessary to keep the analysis tractable and focused.
> Within the clearly stated scope of our theoretical model and assumptions, we have provided rigorous explanations of length generalization. While extending these results beyond our current setting would be valuable future work, the primary goal of this paper is to provide some novel insights from a theoretical perspective. We hope that our answers to your later questions can explain to you why our setting shares the structural similarities with real-world CoT performed by actual language models, and how to interpret our idealized results to shed light on practical applications.
>
> **Q2.** It is unclear whether the results truly apply to CoT, given its different sequence structure and attention patterns.
>
> **A2.**  Through our technical construction, we aim to create a setting where the structure of step-by-step reasoning is preserved and is amenable to theoretical analysis. While it may appear different from natural CoT traces on the surface, we emphasize that our design abstracts two essential components of real-world CoT reasoning:
> 1. Each CoT step typically focuses on a localized subproblem, even when the overall task involves long-range dependencies. We represent this structure using a **clause-based data format**, which allows us to formalize how reasoning steps are represented, sampled, and generated during training.
> 2. Long-range dependencies in CoT are often handled by the retrieval function of attention. This is an important topic of study in both theoretical analyses [3,6,7] and practical transformer components such as *induction heads* [8]. In our framework, we build on the LEGO formulation and adopt the "variables" in LEGO as retrieval queries, thereby enabling transformers to attend to relevant context.
>
> In summary, while our problems are abstract examples of chain-of-thought reasoning, they preserve key structural patterns observed in real-world reasoning traces. We hope these abstractions can offer insights into how attention mechanisms may function in practical CoT models.
>
> **Q3.** The model ties query/key projections and omits value projections. Why isn't this assumption explicitly stated, given it deviates from standard transformer architectures?
>
> **A3**. Thanks for the question. We adopted a quite common simplification in existing theoretical works of transformer training dynamics due to technical tractability [1-5]. Our assumptions delegated all the functionalities of attention to a single attention weight Q, which leaves room to analyze the attention pattern between different tokens while drastically simplifying the analysis because we dropped many interplays between different weight updates. In fact, one can view our setting as we adopted a slightly stronger attention layer, since now the $Q^TK$ is full rank and the value matrix can be thought of as absorbed into the MLP. Thank you for highlighting this point, and we will explicitly include additional references and clarify this assumption to better justify our architectural choices in the revised manuscript.
>
> **Q4.**  Assumption 3.4 appears to be very strong
>
> **A4.** As we pointed out in line 226 in the paper, setting certain parts of the weight matrix to zero aligns with common practices in existing theoretical analyses of transformer training dynamics [1-5]. Technically, for our task, only the blocks $Q_{4,3}$ and $Q_{4,4}$  play a meaningful role in learning: $Q_{4,3}$ is important for in-context retrieval, while $Q_{4,4}$ puts weights on the current state. The remaining components either receive no gradient or capture task-irrelevant biases that complicate the analysis. The delicate balance between these two blocks, and their interaction with irrelevant tokens in the context, is the major technical challenge in our attention dynamics analysis. In fact, some existing works [1-2] assume even stronger simplifications, such as being a scalar multiple of the identity matrix, with only a single learnable parameter, for similar reasons. Even though we have made some compromises in the setting to allow tractable analysis, we hope that our methods can offer some insights on the reasoning ability of general language models. In fact, our result gives a clear example of how transformers can learn to implement the induction head [9], showing that our setting can reproduce certain results on the inner workings of transformers studied in prior works.
>
> **Q5.** How well do the results extend to RoPE or other positional encodings where clause boundaries are not explicitly separated
>
> **A5**. As discussed in Q2, we believe that achieving length generalization beyond the training context requires the model to overcome certain proximity biases. However, we do not aim to discredit the importance of locality that is better captured by other PE such as RoPE, as our clause-based structure actually absorbed all the local correlations to make the analysis possible.  Regarding the extension, we conjecture that currently used positional encodings like RoPE or ALiBi are inherently limited in their length generalizability, due to their position-dependent nature (either absolute or relative), such as shown in some empirical study of architectural design [9]. Prior theoretical works on CoT expressivity [6, 7] also rely on fixed positional encoding schemes and cannot generalize beyond a fixed problem length. We hope our work could motivate further exploration of architectural designs that can enable transformers to extrapolate beyond training context lengths, whether for CoT reasoning or to solve other problems.
>
> **Q6.**  The experiment could benefit from a non-CoT baseline to display the difference in length generalization empirically.
>
> **A6.** Following the reviewer’s suggestion, we conducted additional experiments with a non-CoT baseline in both the simply transitive (cyclic) and symmetric cases to evaluate length generalization. Specifically, we trained the same one-layer transformer under the identical task setup described in the paper, but without intermediate answer clauses, i.e., directly predicting the final answer. For non-CoT models, we explored eight learning rates: $[5e-6, 1e-5, 2e-5, 5e-5, 1e-4, 2e-4, 5e-4]$, whereas our CoT model was trained with a single learning rate ($1e-4$). Additionally, the CoT model was trained exclusively on LEGO sequences of length 5, while the non-CoT baseline was trained on sequences of lengths 5 and 10.
>
> | Cyclic | 5 | 10 | 20 | 50 | 100 | 150 |
> | :--- | :--- | :--- | :--- | :--- | :--- | :--- |
> | CoT (trained on L=5) | 1 | 1 | 1 | 1 | 0.997 | 0.981 |
> | Best (trained on L=5) | 0.164 | 0.169 | 0.151 | 0.181 | 0.192 | 0.19 |
> | Best (trained on L=10) | 0.168 | 0.184 | 0.166 | 0.179 | 0.174 | 0.179 |
>
> | Symmetric | 5 | 10 | 20 | 50 | 100 | 150 |
> | :--- | :--- | :--- | :--- | :--- | :--- | :--- |
> | CoT  (trained on L=5) |0.999 | 0.998 | 0.991 | 0.989 | 0.965 | 0.922 |
> | Best (trained on L=5) | 0.009 | 0.009 | 0.01 | 0.009 | 0.013 | 0.011 |
> | Best (trained on L=10) | 0.01 | 0.01 | 0.013 | 0.005 | 0.008 | 0.011 |
>
> Results show that the transformer with CoT achieves length generalization in our setting, while the non-CoT baseline does not. We also plan to include additional multi-layer transformer experiments to further investigate this setting.
>
> ---
>
> We thank the reviewer again for your comments. We hope that our responses resolved your concerns. If so, we wonder if the reviewer could kindly consider to increase your score. Certainly, we are more than happy to answer your further questions.
>
> ---
>
> [1] Wang et al. Transformers Provably Learn Sparse Token Selection While Fully-Connected Nets Cannot
>
> [2] Nichani et al. How Transformers Learn Causal Structure with Gradient Descent
>
> [3] Kim et al. Transformers provably solve parity efficiently with a chain of thought
>
> [4] Zhang et al. Trained transformers learn linear models in-context
>
> [5] Kim et al. Transformers learn nonlinear features in context: nonconvex mean-field dynamics on the attention landscape
>
> [6] Feng et al. Towards revealing the mystery behind chain of thought: a theoretical perspective
>
> [7] Li et al. Chain of thought empowers transformers to solve inherently serial problems
>
> [8] Olsson et al. In-context Learning and Induction Heads
>
> [9] Ren et al. Samba: Simple Hybrid State Space Models for Efficient Unlimited Context Language Modeling

---

> ### Comment · Reviewer_kcpH · 2025-08-03
> **Response to Rebuttal**
>
> I thank the authors for their thorough reply to my review, as well as to those of my fellow reviewers. In particular, the response to X31B's review highlighted some gritty details of this paper, such as that the task is indeed NC1-complete in the case of non-solvable groups. I think the changes and clarifications outlined by the authors will strengthen the paper substantially.
>
> There are two things that are still a bit unclear to me:
> 1. While the discussion here has convinced me that the assumptions are justified in the sense that they represent the frontier of what we are currently able to say about the theoretical properties of transformer training dynamics, it is still not clear to me why it is $Q_{3, 4}$ and $Q_{4, 4}$ in particular that allow the model to solve this task. Can the authors expand on this a bit? Is it because state-tracking can be implemented with simple one look-behind operator + one update to the current state per timestep?
> 2. The extended empirical results are very confusing to me. They do not appear to suggest that the non-CoT model fails to length-generalize, but rather that it fails to learn the task at all (since performance is roughly flat, and very low even on in-distribution lengths 5-10). Do the authors have any thoughts on why this might be? The task itself does not appear so difficult that it cannot be solved in-distribution without CoT, but perhaps my intuition is wrong.
>
> Other than these two points I do not have any other requests to make of the authors at this point. I will await the replies of my fellow reviewers before officially updating my score in case new evidence is brought to light, but would at this point consider updating my overall score to 5 - Accept as I believe the discussion has improved the paper both in Quality and in Clarity. (I would encourage the authors to spend more time on producing a set of relevant and high-confidence experiments, but since the primary focus of this submission is theoretical I do not believe it should be a reason for the paper to not be accepted.)

---

> > ### Author Response · Authors · 2025-08-04
> > **Response to Follow-up Questions Part 1/2**
> >
> > We sincerely thank the reviewer for the feedback and the recognition of our contributions. Your positive assessment is deeply appreciated and motivating. Below, we address the two remaining questions:
> >
> >
> > **Q1**. it is still not clear to me why it is $Q_{4,3}$ and $Q_{4,4}$ in particular that allow the model to solve this task. Can the authors expand on this a bit? Is it because state-tracking can be implemented with simple one look-behind operator + one update to the current state per timestep?
> >
> >
> > **A1.** Yes, your intuition is largely correct. As we outlined in the proof sketch (line 292), the transformer learns to implement a state-tracking mechanism where the prediction of $y_{i+1} = g_{i+1}(y_i)$ is achieved by (1) attending to the context clause $Z_{\text{pred}, i+1}$ to extract the function $g_{i+1}$, and (2) attending to the answer clause $Z_{\text{ans}, i}$ to extract the previous value $y_i$. The MLP component is then responsible for computing the one-step reasoning, i.e., applying $g_{i+1}(y_i)$ to produce the output.
> >
> > Technically, these two retrievals are implemented via the attention matrix $Q$, which is trained to locate the shared variable token $x_i$. In particular, the variable token $x_i$ appears as the **third** token in the context clause $Z_{\text{pred}, i+1}$ and as the **fourth** token in the answer clause $Z_{\text{ans}, i}$. The transformer uses the token $x_i$ in $Z_{\text{ans}, i}$ as the query token (position 4 in the clause), and attention matrix $Q$ with indices $Q_{43}$ and $Q_{44}$ are learned to attend to (i) the third token in the context clause (to retrieve $g_{i+1}$) and (ii) the fourth token in the answer clause (to retrieve $y_i$), respectively.
> >
> > These parallel attention paths allow the model to simultaneously retrieve both the function and the current state needed for the update. This mechanism, combined with a one-layer MLP that applies the learned function $g$, enables the model to perform the correct state update at each timestep and thereby solves the task.
> >
> > We believe our mathematical model can be extended to support more complex reasoning tasks, for example, when the state transition graph has a more intricate topology than a simple line, or when the transition dynamics go beyond group operations. In the former case, the model may need to implement more elaborate procedures, such as topological sorting, to determine the correct CoT reasoning steps. This aligns with key elements in the constructive theory of CoT proposed in [1], as well as with empirical findings reported in [2].
> >
> > ---
> >
> > *Reference*:
> >
> > [1] Li et al. Chain of thought empowers transformers to solve inherently serial problems
> >
> > [2] Ye et al. Physics of Language Models: Part 2.1, Grade-School Math and the Hidden Reasoning Process

---

> > > ### Author Response · Authors · 2025-08-04
> > > **Response to Follow-up Questions Part 2/2**
> > >
> > > **Q2**. The confusion about the additional empirical results: they do not appear to suggest that the non-CoT model fails to length-generalize, but rather that it fails to learn the task at all. Do the authors have any thoughts on why this might be?
> > >
> > > **A2**. Thanks for pointing this out! You are correct that the model trained on non-CoT data indeed failed to learn the task, even in-distribution. To ensure this result was not merely due to specific configuration choices, we performed additional hyper-parameter sweeps as a sanity check. We summarize these extended results below:
> > >
> > > - **Simply transitive (Cyclic) case**:
> > >    1. We trained transformers without CoT on the C6 task and observed that the task difficulty significantly increases with sequence length. Specifically, we trained models with varying depths (1, 2, 4, 6, 8, 16, 32 layers) on sequences of lengths 2, 3, 4, and 5. Our results show that for sequences shorter than 5, transformers successfully learned the task, achieving test accuracy greater than 99% even with relatively shallow depth. However, for sequences of length 5, even a 32-layer transformer failed to exceed 18% test accuracy, which is close to the performance of random guessing.
> > > | Sequence Length | Minimum Layers for >99% Accuracy |
> > > | :---: | :---: |
> > > | 2 | 1 (100.00%) |
> > > | 3 | 1 (99.90%) |
> > > | 4 | 12 (100.00%)|
> > > | 5 | >32 layers|
> > >
> > >         The **length generalization** performance is shown in the table below. This is quite similar to the results reported in  [1].
> > >         | Layers | Train Len | Acc @ L2 | Acc @ L3 | Acc @ L4 | Acc @ L5 | Acc @ L6 | Acc @ L7 | Acc @ L8 | Acc @ L9 | Acc @ L10 |
> > >         |:---|:---|:---|:---|:---|:---|:---|:---|:---|:---|:---|
> > >         | 1 | 2 | 1.0000 | 0.1730 | 0.1550 | 0.1480 | 0.2000 | 0.1620 | 0.1720 | 0.1550 | 0.1810 |
> > >         | 1 | 3 | 0.0770 | 1.0000 | 0.1520 | 0.1760 | 0.1820 | 0.1750 | 0.1520 | 0.1450 | 0.1480 |
> > >         | 12 | 4 | 0.1750 | 0.0870 | 1.0000 | 0.0700 | 0.1110 | 0.1380 | 0.1750 | 0.2090 | 0.1600 |
> > >
> > >      2. Since we aim to compare against the best-performing non-CoT models, we took additional steps to optimize their performance. Specifically, we trained transformers with depths of 2, 4, 6, and 8 layers on a mixed-length problem distribution (lengths 2, 3, 4, and 5). Under this mixed training regime, the models successfully achieved test accuracy exceeding 99% on all sequence lengths up to 5. Nevertheless, these models still failed to generalize effectively to longer sequences (length >= 6). We summarize these results in the following table.
> > >         | Layers | Acc @ L1    |Acc @ L2    | Acc @ L3    | Acc @ L4    | Acc @ L5    | Acc @ L6    | Acc @ L7    | Acc @ L8    | Acc @ L9    | Acc @ L10   |
> > >         | :---:  | :---: | :---: | :---: | :---: | :---: | :---: | :---: | :---: | :---: | :---: |
> > >         | 2      | 0.999 | 1.000 | 0.996 | 0.990 | 0.998 | 0.870 | 0.172 | 0.059 | 0.240 | 0.247 |
> > >         | 4      | 1.000 | 0.999 | 0.998 | 0.989 | 0.993 | 0.940 | 0.609 | 0.379 | 0.279 | 0.286 |
> > >         | 6      | 1.000 | 0.999 | 0.988 | 0.973 | 0.990 | 0.950 | 0.727 | 0.391 | 0.226 | 0.214 |
> > >         | 8      | 1.000 | 1.000 | 0.999 | 0.999 | 0.995 | 0.931 | 0.717 | 0.518 | 0.341 | 0.232 |
> > > - **Symmetry case**: For group S5, achieving successful reasoning without CoT is significantly more challenging. In fact, we only obtained nontrivial performance when the models were trained on sequences of length 2. For sequences longer than length 2, extensive hyperparameter tuning may be required, which we are currently unable to pursue comprehensively. Below, we present the length generalization performance of a model trained on the S5 task with sequence length L=2.
> > >
> > >   |   Layers |   Train Len |   Acc @ 2 |   Acc @ 3 |   Acc @ 4 |   Acc @ 5 |   Acc @ 6 |   Acc @ 7 |   Acc @ 8 |   Acc @ 9 |   Acc @ 10 |
> > >   |---------:|------------:|----------:|----------:|----------:|----------:|----------:|----------:|----------:|----------:|-----------:|
> > >   |        1 |           2 |     0.021 |     0.004 |     0.01  |     0.005 |     0.004 |     0.008 |     0.013 |     0.01  |      0.006 |
> > >   |        2 |           2 |     0.998 |     0.002 |     0.014 |     0.004 |     0.009 |     0.009 |     0.012 |     0.012 |      0.015 |
> > >   |        4 |           2 |     0.998 |     0.005 |     0.017 |     0.007 |     0.011 |     0.01  |     0.007 |     0.007 |      0.013 |
> > >
> > > We fully acknowledge the importance of empirical validation in complementing the theoretical contributions of our work, and we plan to further develop and refine the experimental component in future research.  We sincerely thank the reviewer once again for the constructive feedback and for the positive assessment of our contributions.
> > >
> > > ---
> > >
> > > *Rererence:*
> > >
> > > [1] Liu et al. Transformers learn shortcuts to automata.

---

> > > > ### Comment · Reviewer_kcpH · 2025-08-04
> > > > **Response to follow-ups**
> > > >
> > > > I thank the authors for their additional clarifications.  The empirical findings for the mixed-length training mode in the simple transitive case look more like what one would expect from the theory. It would be reassuring to have seen similar results in the S5 case as well, but at this point I don't think this would make or break the motivation of the paper either way since any negative findings could always be chalked up to a lack of tuning the models appropriately.
> > > >
> > > > I do not have any further questions for the authors at this time.

---

### Official Review · Reviewer_X31B · 2025-07-02

**Clarity:** 3
**Significance:** 2
**Originality:** 2
**Rating:** 4
**Confidence:** 3

**Summary:**

This paper presents a theoretical analysis of a one-layer transformer's ability to learn Chain-of-Thought (CoT) reasoning via gradient descent. Focusing on the synthetic "state tracking" task, the authors prove that the model can learn to solve instances of this problem that belong to the complexity class NC1. The work further investigates length generalization, showing that for simple algebraic structures (cyclic groups), the learned solution generalizes to longer sequences. For more complex structures (symmetric groups), where direct generalization is limited, the authors propose a "recursive self-training" curriculum and prove that it allows the model to achieve similar polynomial-length generalization.

**Questions:**

1. Could you elaborate on the role of the smoothed ReLU in your theoretical analysis? Are there specific properties of this function (e.g., its behavior for negative inputs or its specific polynomial form) that are essential for the proof to go through, as opposed to other common smooth activations?
2. In the recursive self-training curriculum (Algorithm 2), new data is generated using the model's greedy output (Definition 4.3). From a theoretical standpoint, what challenges would arise if stochastic sampling (e.g., with a non-zero temperature) were used instead of greedy decoding to generate the self-training data? Would the convergence guarantees for length generalization still hold?
3. Theorem 4.2 provides an accuracy of 1−O(2^{−k/poly(d)}) for the bootstrapped model. Could you provide more intuition on how errors might propagate or accumulate as the reasoning length $L$ grows to be a large polynomial in $d$? Does the model become more susceptible to single-step failures as the chain lengthens, and does the theory provide a way to bound this cumulative error?

**Ethical Concerns:**

["NO or VERY MINOR ethics concerns only"]

**Final Justification:**

After the discussion with the authors, most of my concerns have been addressed. I will update my score to weak-accept to reflect this.

**Limitations:**

I do not think that equaling limitations with assumptions makes sense. I suggest the reader to position their work in a more objective way.

**Paper Formatting Concerns:**

NaN

**Quality:**

3

**Strengths And Weaknesses:**

### Strengths

- **Addresses a Fundamental Question:** The paper tackles a central and challenging question in modern AI theory: whether the reasoning capabilities observed in large language models can be provably learned via optimization, moving beyond expressivity results alone.
- **Detailed Technical Analysis:** The paper provides a detailed and technically involved proof for its specific setting, demonstrating a mechanism by which attention concentration can emerge from a staged training process to solve a sequential task.
- **Formalization of Self-Training:** The theoretical formalization and proof for the "recursive self-training" curriculum is a novel contribution. It provides a potential mathematical explanation for the empirical success of self-improvement methods that have been observed in practice.

### Weaknesses

- While I completely acknowledge the necessity of studying why CoT is helpful from an optimization perspective, I think that the single special task is not representative enough for demonstrating the underlying mechanism why models can learn to leverage CoT. Specifically, the authors use the state tracking task in NC1 for demonstration, but they do not explicitly point out that whether state tracking is hard and representative enough to distinguish  TC0 and NC1. For instance, is it NC1-complete? In your citation [24], the paper prove their theorems on differnt tasks including a whole class of dynamic programming problems, making their results universal and convincing. Without showing that the theorem holds for a general class of tasks (rather than hold specifically for this task because of its special structure), I do not think that "explaining why CoT can be learn from an optimization perspective" can be claimed.
- The focus on one-layer transformer makes the optimization perspective more or less similar to a construction (expressivity perspective), as Alg 2 constructes optimization in a manual and step-by-step way for different layers. Since the authors try to go beyond expressivity, and since, in theory, a two layer transformer is more expressive, I think the authors should also prove, at least more generally, that a two / general layers of transformers can also be optimized to learn CoT well. Otherwise, the whole proof for optimization seems to be not generalizable, making the theorems less convincing.
- To the best of my knowledge, your citation [24] proposes "transformers could simulate polynomial-size circuits using poly(L) steps of CoT" earlier than [25]. What is the reason that you choose to base on a later work?

---

> ### Author Rebuttal · Authors · 2025-07-31
>
> We thank the reviewer for the time and thoughtful comments on our work. Below we address your concerns.
>
> **Q1**: The single special task is not representative enough
>
> **A1**:
> 1. We first emphasize that state tracking over group operations has been widely used as a benchmark for separating architectural expressivity in seminal works [1–4]. More importantly, state tracking problems that distinguish TC0 from NC1 or beyond have been used to provably separate not only different neural architectures, but also the expressivity of transformers with v.s. without CoT reasoning [5]. This makes state tracking a meaningful and theoretically grounded testbed for studying the capabilities of transformers, either using CoT or not. The task is also widely adopted in empirical studies of neural reasoning [6–7].
> 2. Importantly, it is a well-established theoretical fact that state tracking over non-solvable groups (e.g., $S_5$) is NC1-complete [1]. We highlight this in direct response to the reviewer’s question, as it is a standard and widely known result in circuit complexity.
> 3. Prior theoretical studies on CoT [8–10] necessarily focus on simple, hand-crafted tasks within TC0 (e.g., parity, linear regression), due to the inherent difficulty of analyzing optimization dynamics. In contrast, our work advances this line of research by studying both TC0- and NC1-level CoT tasks, and by providing the first convergence guarantees in these more expressive settings without overclaiming generality beyond this well-motivated domain.
>
> **Q2.** The focus on one-layer transformers makes the optimization perspective more or less similar to a construction
>
> **A2. We believe there may be a misunderstanding regarding our optimization analysis.**  Our work is fundamentally different from and significantly more challenging than the expressivity argument the reviewer suggests.
> 1. This confusion likely stems from prior optimization-based CoT theory [8–10], which focuses on simple tasks (mostly the parity problem in TC0) where a single gradient step closely mimics a hand-crafted construction. In contrast, our setting requires studying the **multi-phase training dynamics** of both MLP and attention weights across many gradient steps, and there is no straight path from initialized weights to the optimal weights. The features learned in the network go through sophisticated trajectories that we cannot control and characterize completely but can only guarantee its convergence on a high level, **which is very different from construction-style arguments**. Thus this highly nontrivial learning process is nothing like expressivity type construction.
> 2. The reason we introduce a curriculum-style schedule (Alg 2) that separates the learning of MLP (focused on one-step group actions) from the learning of attention (enabling length generalization) is to manage the complexity of the intertwined dynamics between different parts of the network. **This separation does not simplify the learning process into a staged construction**. Both stages still undergo complex multi-phase optimization, and our analysis must carefully track how intermediate solutions influence subsequent stages. This level of coupling introduces deep technical challenges far beyond what construction-based proofs address.
> 3. The implication that one-layer transformers are “simple” to analyze is also misleading. The optimization landscape remains highly nontrivial even in this setting [11–13]. In fact, one-layer models likely represent the **current theoretical frontier** for understanding transformer training dynamics. Extending such results to deeper models, without strong layer-wise assumptions, is out of reach for current research.
>
> Prior CoT learning theory has only addressed very simple tasks such as parity with no results on even cyclic group reasoning or length generalization. To our knowledge, we are the first to bridge the gap between the theoretical expressivity and learnability for transformers with CoT, and to provide a length generalization guarantees that go significantly beyond what previous work achieves. Especially since most prior expressivity results rely on absolute positional encoding and thus cannot length generalize.
>
> **Q3.** What is the reason that you choose to base on a later work [25] rather than [24]?
>
> **A3.** We clarify that prior works on expressivity mainly provide background context rather than form the main foundation of our study.  While  [24] was the first to show that transformers with CoT can go beyond TC0 in expressivity, their results do not establish a separation beyond NC1, as the CoT expressivity lower bound is only NC1-hard, even though the tasks may include P-complete problems. We refer more often to [25] because it offers a more comprehensive and aligned characterization of the complexity classes for transformers with and without CoT. That said, we note [25] is not the only work that proves such results, for example,  [1] also establishes similar findings, albeit with a slightly weaker expressivity upper bound.
>
> **Q4.** The role of the smoothed ReLU in your theoretical analysis
>
> **A4.** The smoothed ReLU is chosen due to two key properties that are essential for our optimization analysis:
> 1. Its polynomial smoothness around zero is necessary for applying the tensor power method, which plays a central role in proving MLP feature emergence, as shown in prior works [14–16].
> 2. The symmetry around zero serves as a soft barrier that prevents some pre-activations from drifting too far into the negative region. This helps ensure convergence to a clean and interpretable set of features. We adopt this design specifically to enable our length-generalization result.
>
> In summary, the use of smoothed ReLU makes the optimization analysis more tractable, allowing us to show that the transformer learns clean features for one-step state tracking. This enables us to focus on analyzing length extension and extrapolation, which is the core contribution of this work.
>
> **Q5.** Challenges of using stochastic sampling (e.g., non-zero temperature) instead of greedy decoding for generating self-training data?
>
> **A5.** The main challenge lies more in the clarity and tractability of the proof rather than in any fundamental technical obstacle. In fact, even with non-zero temperature, at the end of each training length $2^k$ for $k \in N^+$, the probability of generating an incorrect prediction has already fallen below 1/d with high probability over the dataset. A more involved analysis would show that the noise introduced by occasional incorrect predictions due to stochastic sampling does not significantly affect the learning of the attention weights. Sequences with slightly higher error rates occur with exponentially small probability, and thus have a negligible impact on the overall attention gradients. However, to analyze such gradients in detail would require extensive case-by-case discussion, which would make the proof significantly more cumbersome without offering fundamentally new insights.
>
> **Q6.** Intuition on how errors might propagate or accumulate as the reasoning length grows to be a large polynomial in d
>
> **A6.** This is a great question, and we agree that a clearer explanation of the limits of length extrapolation should be included in our paper. At a very high level, as context length grows beyond the training length, attention to important clauses becomes diluted, and the influence of incorrect group operations can accumulate for CoT prediction. If the total attention to misleading tokens becomes significant, the model’s prediction can deteriorate. In the simply transitive action case, the correct operation typically receives $1 - 1/d^c$ attention, while all others get $1/d^c$ in total. Thus, one would need roughly $L \cdot d^c$ incorrect operations in the context to confuse the model, allowing extrapolation to polynomially longer lengths. Beyond length $\gg |Y| \cdot L \cdot d^c$, predictions approach uniform. In the symmetric group action case, the learned representations are less balanced, and error arises for length $> C \cdot L$ with a small constant $C < 10$. Beyond that limit, the prediction of transformers for values $y \in \mathcal{Y}$ can quickly become uniformly random.
>
> ---
>
> We thank the reviewer again for your comments. We hope that our responses resolved your concerns. If so, we wonder if the reviewer could kindly consider to increase your score. Certainly, we are more than happy to answer your further questions.
>
> ---
> References:
>
> [1] Merrill et al. The expressive power of transformers with chain of thought
>
> [2] Merrill et al.,  A Little Depth Goes a Long Way: The Expressive Power of Log-Depth Transformers
>
> [3] Merrill et al. The Illusion of State in State-Space Models
>
> [4] Liu et al. Transformers learn shortcuts to automata
>
> [5] Li et al. Chain of thought empowers transformers to solve inherently serial problems
>
> [6] Kim et al. Entity tracking in language models
>
> [7] Toshniwal et al. Chess as a Testbed for Language Model State Tracking
>
> [8]  Huang et al. Transformers learn to implement multi-step gradient descent with chain of thought
>
> [9]  Kim et al.  Transformers provably solve parity efficiently with a chain of thought
>
> [10] Wen et al. From sparse dependence to sparse attention: unveiling how chain-of-thought enhances transformer sample efficiency
>
> [11] Wang et al. Transformers Provably Learn Sparse Token Selection While Fully-Connected Nets Cannot
>
> [12] Zhang et al. Trained transformers learn linear models in-context
>
> [13]Li et al.  A Theoretical Understanding of shallow Vision Transformers: Learning, Generalization, and Sample Complexity
>
> [14] Cao et al. Benign Overfitting in Two-layer Convolutional Neural Networks
>
> [15] Allen-Zhu et al., Towards understanding ensemble, knowledge distillation and self-distillation in deep learning
>
> [16] Chen et al. Towards understanding the mixture-of-experts layer in deep learning

---

> > ### Comment · Reviewer_X31B · 2025-08-07
> >
> > The reviewer thanks the authors for the response. Most of my concerns are addressed. I will update my rating accordingly.

---

> ### Author Response · Authors · 2025-08-04
>
> Dear Reviewer X31B,
>
> As the author-reviewer discussion period will end soon, we would like to check whether our responses have properly addressed your concerns? If so, could you please kindly consider increasing your initial score accordingly? Certainly, we are more than happy to answer your further questions.
>
> Thank you for your time and effort in reviewing our work!
>
> Best Regards,
>
> Authors

---

> ### Author Response · Authors · 2025-08-06
> **Your prompt response is highly appreciated**
>
> Dear Reviewer X31B,
>
> As the author-reviewer discussion period ends soon, we would greatly appreciate it if your could check our response promptly. If our response resolves your concerns, we kindly ask you to consider raising the rating of our work. We are also more than happy to answer your further questions. Thank you very much for your time and efforts!
>
> Many thanks,
>
> Authors

---

### Note · Authors · 2025-08-11

We sincerely thank the reviewers and the AC  for their time, thoughtful comments, and constructive feedback throughout the review and discussion process.

Our paper provides a theoretical study of how the CoT reasoning ability of transformers emerges from the perspective of optimization and training dynamics. Specifically, we present the **first convergence guarantees for both TC0- and NC1-level CoT tasks**, which go beyond prior results that focus only on simple TC0 tasks like parity. More importantly, we are the first to establish convergence guarantees that demonstrate not only the transformer’s ability to solve these tasks but, crucially, its capacity for **length generalization**.


We are encouraged that most reviewers found our contributions meaningful and our rebuttal satisfactory. We would like to briefly address one remaining point raised by Reviewer vE9B: "I think omitting positional embeddings is not helpful in increasing the impact of this work.” While we provided a detailed clarification in **A3**, the reviewer remains unconvinced. We respectfully believe our response fully addresses this concern. In particular, we noted that it is widely observed empirically that standard positional embeddings hinder length generalization due to inherent inductive biases favoring local positions. In contrast, our theoretical analysis supports an alternative design without positional embeddings that provably achieves length generalization. This perspective not only helps explain why current empirical setups often fail to generalize to longer inputs, but also offers guidance for future architectural design. This concern appears to reflect a difference in viewpoint, rather than a flaw in the technical correctness or significance of our results. We also note that Reviewer kcpH raised a similar point as a question (**Q5**) rather than a concern, and did not view it as undermining the main contributions of the paper.

We once again thank the reviewers for their constructive input. We hope the AC will consider the resolution of key concerns across the reviews. We believe the paper offers timely and rigorous contributions to the theoretical understanding of transformer reasoning capabilities.

---

### Decision · Program_Chairs · 2025-09-17

**Decision:**

Accept (poster)

**Comment:**

This paper provides a theoretical analysis of how a one-layer transformer learns to perform CoT reasoning via gradient descent. The analysis focuses on a synthetic "state tracking" task over group operations.

Overall, reviewers agree that the paper is well-written; the problem being addressed is a fundamental problem; and the technical analysis is rigorous and solid. Reviewers questioned about the representativeness of the state tracking task, the simplicity of the one-layer transformer, the lack of non-CoT baseline for generation, and assumptions of no PEs. I think the authors did a very good job in addressing these concerns.

I think this is a nice piece of work with solid theoretical contributions and therefore I recommend to accept this paper.